# Eliminating Sharp Minima from SGD with Truncated Heavy-tailed Noise

**Xingyu Wang**[1]    **Sewoong Oh**[2]    **Chang-Han Rhee**[1]
[1]Northwestern University,    [2]University of Washington
`xingyuwang2017@u.northwestern.edu`

## Abstract

The empirical success of deep learning is often attributed to SGD's mysterious ability to avoid sharp local minima in the loss landscape, as sharp minima are known to lead to poor generalization. Recently, evidence of *heavy-tailed* gradient noise was reported in many deep learning tasks, and it was shown in (Şimşekli et al., 2019a;b) that SGD can *escape* sharp local minima under the presence of such heavy-tailed gradient noise, providing a partial solution to the mystery. In this work, we analyze a popular variant of SGD where gradients are truncated above a fixed threshold. We show that it achieves a stronger notion of avoiding sharp minima: it can effectively *eliminate* sharp local minima entirely from its training trajectory. We characterize the dynamics of truncated SGD driven by heavy-tailed noises. First, we show that the truncation threshold and width of the attraction field dictate the order of the first exit time from the associated local minimum. Moreover, when the objective function satisfies appropriate structural conditions, we prove that as the learning rate decreases the dynamics of the heavy-tailed truncated SGD closely resemble those of a continuous-time Markov chain that never visits any sharp minima. Real data experiments on deep learning confirm our theoretical prediction that heavy-tailed SGD with gradient clipping finds a "flatter" local minima and achieves better generalization.

## 1 Introduction

Stochastic gradient descent (SGD) and its variants have seen unprecedented empirical successes in training deep neural networks. The training of deep neural networks is typically posed as a non-convex optimization problem, and even without explicit regularization the solutions obtained by SGD often perform surprisingly well on test data. Such an unexpected generalization performance of SGD in deep neural networks are often attributed to SGD's ability to avoid *sharp local minima*[1] in the loss landscape, which tends to lead to poor generalization (Hochreiter & Schmidhuber, 1997; Keskar et al., 2016; Li et al., 2018b; Jiang et al., 2019); see Appendix D for more details. Despite significant efforts to explain such phenomena theoretically, understanding how SGD manages to avoid sharp local minima and end up with flat local minima within a realistic training time still remains as a central mystery of deep learning. [2] Recently, the heavy-tailed dynamics of SGD received significant attention, and it was suggested that the heavy tails in the stochastic gradients may be a key ingredient that facilitates SGD's escape from sharp local minima: for example, Şimşekli et al. (2019a;b) report the empirical evidence of heavy-tails in stochastic gradient noise in popular deep learning architectures (see also (Hodgkinson & Mahoney, 2020; Srinivasan et al., 2021; Garg et al., 2021)) and show that SGD can escape sharp local minima in polynomial time under the presence of the heavy-tailed gradient noise. More specifically, they view heavy-tailed SGDs as discrete approximations of Lévy driven Langevin equations and argue that the amount of time SGD trajectory spends in each local minimum is proportional to the width of the associated minimum according to the metastability theory (Pavlyukevich, 2007; Imkeller et al., 2010a;b) for such heavy-tailed processes.

---

[1]We use the terminology sharpness in a broad sense; we refer to Appendix C for a detailed discussion.

[2]To see a detailed discussion on existing results on selection of local minima from the stability perspective and the novelty of our analysis, see Appendix E.

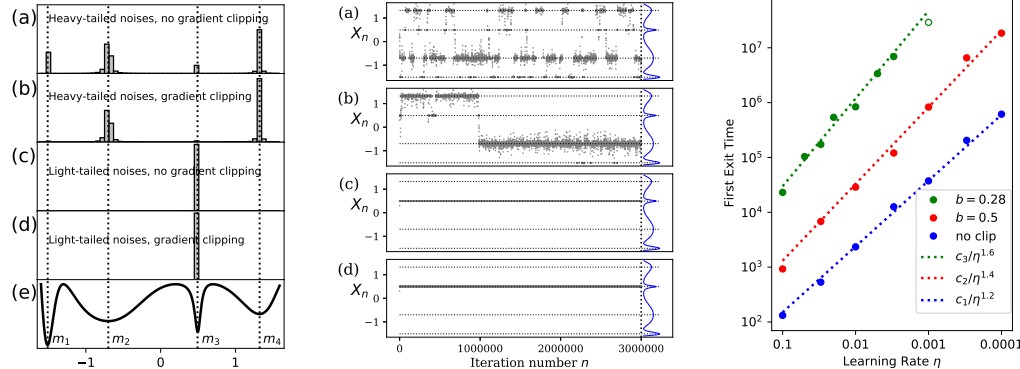

Figure 1: **(Left)** Histograms of the locations visited by SGD. With truncated heavy-tailed noises, SGD hardly ever visits the two sharp minima $m_1$ and $m_3$. The objective function $f$ is plotted at the bottom, and dashed lines are added as references for the locations of local minima. **(Middle)** Typical trajectories of SGD in different cases: (a) Heavy-tailed noises, no gradient clipping; (b) Heavy-tailed noises, gradient clipping at $b = 0.5$; (c) Light-tailed noises, no gradient clipping; (d) Light-tailed noises, gradient clipping at $b = 0.5$. The objective function $f$ is plotted at the right of each figure, and dashed lines are added as references for locations of the local minima. **(Right)** First Exit Time from $\Omega_2 = (-1.3, 0.2)$. Each dot represents the average of 20 samples of first exit time. Each dahsed line shows a polynomial function $c_i/\eta^\beta$ where $\beta$ is predicted by Theorem 1 and $c_i$ is chosen to fit the dots. The non-solid green dot indicates that for some of the 20 samples of the termination threshold $5 \times 10^7$ was reached, and hence, it is an underestimation. Results in **(Left)** and **(Middle)** are obtained under learning rate $\eta = 0.001$ and initial condition $X_0 = 0.3$.

In this paper, we study the global dynamics and long-run behavior of heavy-tailed SGD and its practical variant in depth. While in full generality the structure of gradient noises in SGD is state-dependent, in this work we focus on the role of noise magnitude and analyze the setting where each SGD update is perturbed by iid heavy-tailed noise. In particular, we consider an adaptive version of SGD, where the stochastic gradient is truncated above a fixed threshold. Such truncation scheme is often called *gradient clipping* and employed as default in various contexts (Engstrom et al., 2020; Merity et al., 2018; Graves, 2013; Pascanu et al., 2013; Zhang et al., 2020; Gorbunov et al., 2020). We uncover a rich mathematical structure in the global dynamics of SGD under this scheme and prove that the asymptotic behavior of such SGD is fundamentally different from that of the pure form of SGD: in particular, under a suitable structural condition on the geometry of the loss landscape, gradient clipping *completely eliminates sharp minima from the trajectory of SGDs*. This provides a critical insight into how heavy-tailed dynamics of SGD can be utilized to find a local minimum that generalizes better.

Figure 1 (Left, Middle) clearly illustrates these points with the histograms of the sample trajectories of SGDs. Note first that SGDs with light-tailed gradient noise—(c) and (d) of Figure 1 (Left, Middle)— never manages to escape a (sharp) minimum regardless of gradient clipping. In contrast, SGDs with heavy-tailed gradient noise—(a) and (b) of Figure 1 (Left, Middle)—easily escapes from local minima. Moreover, there is a clear difference between SGDs with gradient clipping and without gradient clipping. In (a) of Figure 1 (Left), SGD without gradient clipping spends a significant amount of time at each of all four local minima ($\{m_1, m_2, m_3, m_4\}$), although it spends more time around the wide ones ($\{m_2, m_4\}$) than the sharp ones ($\{m_1, m_3\}$). On the other hand, in (b) of Figure 1 (Left), SGD with gradient clipping not only escapes from local minima but also avoids sharp minima ($\{m_1, m_3\}$) almost completely. This means that after we run SGD for long enough (more precisely, the required run length $t/\eta^\beta$ is of polynomial order; see Theorem 2), it is almost guaranteed that it won't be at a sharp minimum, effectively eliminating sharp minima from its training trajectories.

We also propose a novel computational strategy that takes advantage of our newly discovered global dynamics of the heavy-tailed SGD. While the evidence of heavy tails were reported in many deep learning tasks (Şimşekli et al., 2019b;a; Garg et al., 2021; Gurbuzbalaban et al., 2020; Hodgkinson

& Mahoney, 2020; Nguyen et al., 2019; Mahoney & Martin, 2019; Srinivasan et al., 2021; Zhang et al., 2020), there seem to be plenty of deep learning contexts where the stochastic gradient noises are light-tailed (Panigrahi et al., 2019) as well. [3] Guided by our new theory, we propose an algorithm that injects heavy-tails to SGD by inflating the tail distribution of the gradient noise and facilitating the discovery of a local minimum that generalizes better. Our experiments with image classification tasks, reported in Tables 1 and 2, illustrate that the tail-inflation strategy we propose here can indeed improve the generalization performance of the SGD as predicted by our theory.

The rest of the paper is organized as follows. Section 2 formulates the problem setting and characterizes the global dynamics of the SGD driven by heavy-tailed noises. Section 3 presents numerical experiments that confirm our theory. Section 4 proposes a new algorithm that artificially injects heavy tailed gradient noise in actual deep learning tasks and demonstrate the improved performance.

**Technical Contributions:** 1) We rigorously characterize the global behavior of the heavy-tailed SGD with gradient clipping. We first focus on the case where the loss function is in $\mathbb{R}^1$ with some simplifying assumptions on its geometry. Even with such assumptions, our theorem involves substantial technical challenges since the traditional tools for analyzing SGD fail in our context due to the adaptive nature of its dynamics and non-Gaussian distributional assumptions. For example, while the unclipped pure SGD can be analyzed by partitioning its trajectory at arrival times of large noises (as in Pavlyukevich (2005) and Imkeller et al. (2010a)), such an approach falls short in our context. Instead, we developed a set of delicate arguments for dealing with SGD's (near) regeneration structure and the return times to the local minima, as well as controlling the probability of atypical scenarios that would not arise in the unclipped case. Moreover, as evidenced by our $\mathbb{R}^d$ results in Appendix I, the approach developed here is critical in extending the analysis to general loss landscapes.

2) We propose a novel computational strategy for improving the generalization performance of SGD by carefully injecting heavy-tailed noise. We test the proposed algorithm with deep learning tasks and demonstrate its superiority with an ablation study. This also suggests that the key phenomenon we characterize in our theory— elimination of sharp local minima—manifests in real-world tasks.

## 2 THEORETICAL RESULTS

This section characterizes the global dynamics of SGD with gradient clipping when applied to a non-convex objective function $f$. In Section 2.1 and 2.2, we make the following assumptions for the sake of the simplicity of analysis. However, as our multidimensional result in Section 2.3 and the experiments in Section 3 and 4 suggest, the gist of the phenomena we analyze—elimination of sharp local minima—persists in general contexts where the domain of $f$ is multi-dimensional, and the stationary points are not necessarily strict local optima separated from one another.

**Assumption 1.** *Let $f : \mathbb{R} \to \mathbb{R}$ be a $\mathcal{C}^2$ function. There exist a positive real $L > 0$, a positive integer $n_{min}$ and an ordered sequence of real numbers $m_1, s_1, m_2, s_2, \cdots, s_{n_{min}-1}, m_{n_{min}}$ such that (1) $-L < m_1 < s_1 < m_2 < s_2 < \cdots < s_{n_{min}-1} < m_{n_{min}} < L$; (2) $f'(x) = 0$ iff $x \in \{m_1, s_1, \cdots, s_{n_{min}-1}, m_{n_{min}}\}$; (3) For any $x \in \{m_1, m_2, \cdots, m_{n_{min}}\}$, $f''(x) > 0$; (4) For any $x \in \{s_1, s_2, \cdots, s_{n_{min}-1}\}$, $f''(x) < 0$.*

As illustrated in Figure 2 (Left), the assumption above requires that $f$ has finitely many local minima (to be specific, the count is $n_{\min}$), all of which contained in some compact domain $[-L, L]$. Moreover, the points $s_1, \cdots, s_{n_{\min}-1}$ naturally partition the entire real line into different regions $\Omega_i = (s_{i-1}, s_i)$ (here we adopt the convention that $s_0 = -\infty, s_{n_{\min}} = +\infty$). We call each region $\Omega_i$ the **attraction field** of the local minimum $m_i$, as the gradient flow in $\Omega_i$ always points to $m_i$.

Throughout the optimization procedure, given any location $x \in \mathbb{R}$ we assume that we have access to the noisy estimator $f'(x) - Z_n$ of the true gradient $f'(x)$, and $f'(x)$ itself is difficult to evaluate. Specifically, in this work we are interested in the case where the iid sequence of noises $(Z_n)_{n \geq 1}$ are heavy-tailed. Typically, the heavy-tailed phenomena are captured by the concept of regular variation: for a measurable function $\phi : \mathbb{R}_+ \mapsto \mathbb{R}_+$, we say that $\phi$ is regularly varying at $+\infty$ with index $\beta$ (denoted as $\phi \in \mathcal{RV}_\beta$) if $\lim_{x \to \infty} \phi(tx)/\phi(x) = t^\beta$ for all $t > 0$. For details on the definition and properties of regularly varying functions, see, for example, chapter 2 of Resnick (2007). In this paper,

---

[3]For a detailed comparison to existing works on heavy-tailed phenomena is SGD, see Appendix F.

we work with the following distributional assumption on the gradient noise. Let

$$H_+(x) \triangleq \mathbb{P}(Z_1 > x), \quad H_-(x) \triangleq \mathbb{P}(Z_1 < -x), \quad H(x) \triangleq H_+(x) + H_-(x) = \mathbb{P}(|Z_1| > x).$$

**Assumption 2.** $\mathbb{E}Z_1 = 0$. *Furthermore, there exists some $\alpha \in (1, \infty)$ such that function $H(x)$ is regularly varying (at $+\infty$) with index $-\alpha$. Besides, regarding the positive and negative tail for distribution of the noises, we have*

$$\lim_{x \to \infty} \frac{H_+(x)}{H(x)} = p_+, \ \lim_{x \to \infty} \frac{H_-(x)}{H(x)} = p_- = 1 - p_+$$

*where $p_+$ and $p_-$ are constants in interval $(0, 1)$.*

Roughly speaking, Assumption 2 means that the shape of the tail for the distribution of noises $Z_n$ resembles a polynomial function $x^{-\alpha}$, which is much heavier than the exponential tail of Gaussian distributions. Therefore, large values of $Z_n$ are much more likely to be observed under Assumption 2 compared to the typical Gaussian assumption. The index $\alpha$ of regular variation encodes the heaviness of the tail—the smaller the heavier—and we are assuming that the left and right tails share the same index $\alpha$. The purpose of this simplifying assumption is clarity of presentation, but our $\mathbb{R}^d$ results in Appendix I relax such a condition and allow different regular variation indices in different directions.

Our work concerns a popular variant of SGD where the stochastic gradient is truncated. Specifically, when updating the SGD iterates with a learning rate $\eta > 0$, rather than using the original noisy gradient descent step $\eta(f'(X_n) - Z_n)$, we will truncate it at a threshold $b > 0$ and use $\varphi_b\big(\eta(f'(X_n) - Z_n)\big)$ instead. Here the truncation operator $\varphi.(\cdot)$ is defined as

$$\varphi_c(w) \triangleq w \cdot \min\{1, c/|w|\} \quad \forall w \in \mathbb{R}, c > 0. \tag{1}$$

Besides truncating the stochastic gradient, we also project the SGD into $[-L, L]$ at each iteration; recall that $L$ is the constant in Assumption 1. That is, the main object of our study is the stochastic process $\{X_j^\eta\}_{j \geq 0}$ driven by the following recursion

$$X_j^\eta \triangleq \varphi_L\Big(X_{j-1}^\eta - \varphi_b\big(\eta(f'(X_{j-1}^\eta) - Z_j)\big)\Big). \tag{2}$$

The projection $\varphi_L$ and truncation $\varphi_b$ here are common practices in many learning tasks for the purpose of ensuring that the SGD does not explode and drift to infinity. Besides, the projection also allows us to drop the sophisticated assumptions on the tail behaviors of $f$ that are commonly seen in previous works (see, for instance, the dissipativity conditions in Nguyen et al. (2019)). For technical reasons, we make the following assumption about the truncation threshold $b > 0$. Note that this assumption is a very mild one, as it is obviously satisfied by (Lebesgue) almost every $b > 0$.

**Assumption 3.** *For each $i = 1, 2, \cdots, n_{min}$, $\min\{|s_i - m_i|, |s_{i-1} - m_i|\}/b$ is not an integer.*

## 2.1 FIRST EXIT TIMES

Denote the SGD's first exit time from the attraction field $\Omega_i$ with $\sigma_i(\eta) \triangleq \min\{n \geq 0 : X_n^\eta \notin \Omega_i\}$. In this section, we prove that $\sigma_i(\eta, x)$ converges to an exponential distribution when scaled properly. To characterize such a scaling, we first introduce a few concepts. For each attraction field $\Omega_i$, define (note that $\lceil x \rceil = \min\{n \in \mathbb{Z} : n \geq x\}, \lfloor x \rfloor = \max\{n \in \mathbb{Z} : n \leq x\}$ )

$$r_i \triangleq \min\{|m_i - s_{i-1}|, |s_i - m_i|\}, \quad l_i^* \triangleq \lceil r_i/b \rceil. \tag{3}$$

Note that $l_i^*$'s in fact depend on the the value of gradient clipping threshold $b$ even though this dependency is not highlighted by the notation. Here $r_i$ can be interpreted as the radius or the effective *width* of the attraction field, and $l_i^*$ is the *minimum number of jumps* required to escape $\Omega_i$ when starting from $m_i$. Indeed, the gradient clipping threshold $b$ dictates that no single SGD step can travel more than $b$, and to exit $\Omega_i$ when starting from $m_i$ we can see that at least $\lceil r_i/b \rceil$ steps are required. We can interpret $l_i^*$ as the minimum effort required to exit $\Omega_i$. In this sense, $l_i^*$ is an indicator of the width of the attraction field $\Omega_i$. Theorem 1 states that $l_i^*$ dictates the order of magnitude of the first exit time as well as where the iterates $X_n^\eta$ land on at the first exit time. For each $\Omega_i$, define a scaling function $\lambda_i(\eta) \triangleq H(1/\eta) \left((1/\eta)H(1/\eta)\right)^{l_i^* - 1}$. To stress the initial condition, we write $\mathbb{P}_x$ for the probability law when conditioning on $X_0^\eta = x$, or simply write $X_n^\eta(x)$.

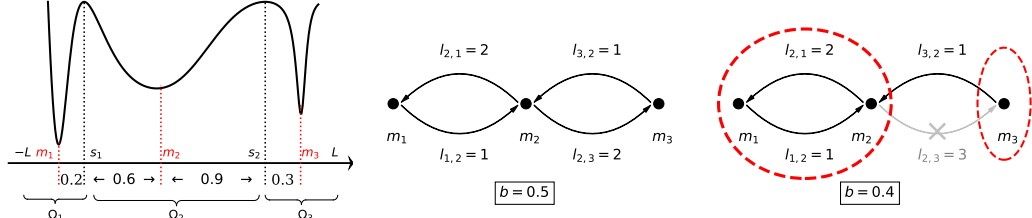

Figure 2: Typical transition graphs $\mathcal{G}$ under different gradient clipping thresholds $b$. (Left) The function $f$ illustrated here has 3 attraction fields. For the second one $\Omega_2 = (s_1, s_2)$, we have $s_2 - m_2 = 0.9, m_2 - s_1 = 0.6$. (Middle) The typical transition graph induced by $b = 0.5$. The entire graph $\mathcal{G}$ is irreducible since all nodes communicate with each other. (Right) The typical transition graph induced by $b = 0.4$. When $b = 0.4$, since $0.6 < 2b$ and $0.9 > 2b$, the SGD can only exit $\Omega_2$ from the left with only 2 jumps if started from $m_2$. Therefore, on the graph $\mathcal{G}$ there are two communication classes: $G_1 = \{m_1, m_2\}, G_2 = \{m_3\}$; $G_1$ is absorbing while $G_2$ is transient.

**Theorem 1.** *Under Assumptions 1-3, there exist constants $q_i > 0 \ \forall i \in \{1, 2, \cdots, n_{min}\}$ and $q_{i,j} \geq 0 \ \forall j \in \{1, 2, \cdots, n_{min}\} \setminus \{i\}$ such that*

*(i) Suppose that $x \in \Omega_k$ for some $k \in \{1, 2, \cdots, n_{min}\}$. Under $\mathbb{P}_x$, the scaled first exit time $q_k \lambda_k(\eta) \sigma_k(\eta)$ converges in distribution to $Exp(1)$ as $\eta \downarrow 0$.*

*(ii) For $k, l \in \{1, 2, \cdots, n_{min}\}$ such that $k \neq l$, we have $\lim_{\eta \to 0} \mathbb{P}_x(X^\eta_{\sigma_k(\eta)} \in \Omega_l) = q_{k,l}/q_k$.*

The proof and discussion are deferred to Appendix G. The constants $q_i, q_{i,j}$ are explicitly identified in terms of the gradient flows perturbed by Pareto jumps in Section C of Appendix. We note here that Theorem 1 implies (i) for $X^\eta_n$ to escape the current attraction field, say $\Omega_i$, it takes $O\big(1/\lambda_i(\eta)\big)$ time, and (ii) the destination is most likely to be reachable within $l^*_i$ jumps from $m_i$.

## 2.2 ELIMINATION OF SMALL ATTRACTION FIELDS

Under proper structural assumptions on the geometry of $f$, the sharp minima of $f$ can be effectively eliminated from the trajectory of heavy-tailed SGD, facilitating the discovery of flat minima. This is somewhat surprising given that gradient clipping mechanism makes the SGD iterates move *slower*. The intuition behind this is that for narrow basins, applying gradient clipping has virtually no effect on the order of exit time; whereas for a wide basin that requires multiple jumps to escape under the clipping scheme, the clipping of gradients significantly slows down the escape and makes SGD stay longer. In other words, gradient clipping only makes SGDs stay longer in the wider (better) basins.

Now, we introduce a few new concepts. Similar to the the minimum number of jumps $l^*_i$ defined in (3), we define the following as the *minimum number of jumps to reach $\Omega_j$ from $m_i$* for any $j \neq i$:

$$l_{i,j} = \begin{cases} \lceil (s_{j-1} - m_i)/b \rceil & \text{if } j > i, \\ \lceil (m_i - s_j)/b \rceil & \text{if } j < i. \end{cases} \quad (4)$$

Recall that Theorem 1 dictates that $X^\eta_n$ is most likely to move out of the current attraction field, say $\Omega_i$, to somewhere else after $O\big(1/\lambda_i(\eta)\big)$ time steps, and the destination is most likely to be reachable within $l^*_i$ jumps from $m_i$. Therefore, the transitions from $\Omega_i$ to $\Omega_j$ can be considered typical if $\Omega_j$ can be reached from $m_i$ with $l^*_i$ jumps—that is, $l_{i,j} = l^*_i$. Now we define the following directed graph that only includes these typical transitions.

**Definition 1** (Typical Transition Graph). *Given a function $f$ satisfying Assumption 1 and gradient clipping threshold $b > 0$ satisfying Assumption 3, a directed graph $\mathcal{G} = (V, E)$ is the corresponding typical transition graph if (1) $V = \{m_1, \cdots, m_{n_{min}}\}$; (2) An edge $(m_i \to m_j)$ is in $E$ iff $l_{i,j} = l^*_i$.*

Naturally, the typical transition graph $\mathcal{G}$ can be decomposed into different communication classes $G_1, \cdots, G_K$ that are mutually exclusive by considering the equivalence relation associated with the existence of the (two-way) paths between $i$ and $j$. More specifically, for $i \neq j$, we say that

$i$ and $j$ communicate if and only if there exists a path $(m_i, m_{k_1}, \cdots, m_{k_n}, m_j)$ as well as a path $(m_j, m_{k'_1}, \cdots, m_{k'_{n'}}, m_i)$ in $\mathcal{G}$; in other words, by travelling through edges on $\mathcal{G}$, $m_i$ can be reached from $m_j$ and $m_j$ can be reached from $m_i$.

We say that a communication class $G$ is *absorbing* if there does not exist any edge $(m_i \to m_j) \in E$ such that $m_i \in G$ and $m_j \notin G$. Otherwise, we say that $G$ is *transient*. In the case that all $m_i$'s communicate with each other on graph $\mathcal{G}$, we say $\mathcal{G}$ is *irreducible*. See Figure 2 (Middle) for the illustration of an irreducible case. When $\mathcal{G}$ is irreducible, we define the set of *largest* attraction fields $M^{\text{large}} \triangleq \{m_i : i = 1, 2, \cdots, n_{\min}, l_i^* = l^{\text{large}}\}$ where $l^{\text{large}} = \max_j l_j^*$; recall that $l_i^*$ characterizes the width of $\Omega_i$. Define the longest time scale $\lambda^{\text{large}}(\eta) = H(1/\eta)((H(1/\eta)/\eta))^{l^{\text{large}}-1}$. Note that this corresponds exactly to the order of the first exit time of the largest attraction fields; see Theorem 1. The following theorem is the main result of this paper.

**Theorem 2.** *Let Assumptions 1-3 hold and assume that the graph $\mathcal{G}$ is **irreducible**. For any $t > 0$, $\beta > 1 + (\alpha - 1)l^{large}$ and $x \in [-L, L]$,*

$$\frac{1}{\lfloor t/\eta^\beta \rfloor} \int_0^{\lfloor t/\eta^\beta \rfloor} \mathbb{1}\Big\{ X_{\lfloor u \rfloor}^\eta(x) \in \bigcup_{j : m_j \notin M^{large}} \Omega_j \Big\} du \to 0 \tag{5}$$

*in probability as $\eta \to 0$.*

The proof is deferred to Appendix H. Here we briefly discuss the implication of the result. Suppose that we terminate the training after a reasonably long time, say, $\lfloor t/\eta^\beta \rfloor$ iterations. Then the random variable that converges to zero in eq. (5) is exactly the proportion of time that $X_n^\eta$ spent in the attraction fields that are not wide. Therefore, by truncating the gradient noise of the heavy-tailed SGD, we can effectively eliminate small attraction fields from its training trajectory. In other words, it is almost guaranteed that SGD is in one of the widest attraction fields after sufficiently long training. Meanwhile, despite the asymptotic nature of Theorem 2, it has been confirmed in our simulation and deep learning experiments that the elimination effect can be observed under typical choices of $\eta$.

Theorem 2 is merely a manifestation of the global dynamics of heavy-tailed SGD. The main messages of the next theorem are: (a) under clipped heavy-tailed noises, the dynamics of $X_n^\eta$ for small $\eta$ closely resemble that of a continuous-time Markov chain; (b) this chain *only* visits local minima of the largest attraction fields of $f$, thus minima in small attraction fields are completely avoided.

**Theorem 3.** *Let $x \in \Omega_i$ for some $i = 1, 2, \cdots, n_{min}$. If Assumptions 1-3 hold and $\mathcal{G}$ is irreducible, then there exist a continuous-time Markov chain $Y$ on $M^{large}$ and a random mapping $\pi$ satisfying*

- $\pi(m) \equiv m$ *if $m \in M^{large}$;*

- $\pi(m)$ *is a random variable that only takes value in $M^{large}$ if $m \notin M^{large}$.*

*such that the scaled process $\{ X_{\lfloor t/\lambda^{large}(\eta) \rfloor}^\eta(x) : t \geq 0 \}$ converges to process $\{ Y_t(\pi(m_i)) : t \geq 0 \}$ in the sense of finite-dimensional distributions: for any positive integer $k$ and any $0 < t_1 < t_2 < \cdots < t_k$, the random vector $\left( X_{\lfloor t_1/\lambda^{large}(\eta) \rfloor}^\eta(x), \cdots, X_{\lfloor t_k/\lambda^{large}(\eta) \rfloor}^\eta(x) \right)$ converges in distribution to $\left( Y_{t_1}(\pi(m_i)), \cdots, Y_{t_k}(\pi(m_i)) \right)$ as $\eta \downarrow 0$.*

In section D of Appendix, we detail the proof, the exact parametrization of the generator matrix of process $Y$, and the distribution of random mapping $\pi(\cdot)$. Here we add some remarks. Intuitively speaking, this result tells us that, regardless of where we initialize the SGD iterates, the dynamics of the clipped heavy-tailed SGD converge to a continuous-time Markov chain avoiding any local minima that is not in the largest attraction fields. Second, under small learning rate $\eta > 0$, if $X_n^\eta(x)$ is initialized at $x \in \Omega_i$ where $\Omega_i$ is NOT a largest attraction field, then SGD will quickly escape $\Omega_i$ and arrive at some $\Omega_j$ that is indeed a largest one—i.e., $m_j \in M^{\text{large}}$; such a transition is so quick that, under time scaling $\lambda^{\text{large}}(\eta)$, it is almost instantaneous as if $X_n^\eta(x)$ is actually initialized randomly at some of the largest attraction fields. This randomness is compressed in the random mapping $\pi$. In Section B, we discuss how our characterization in Theorem 3 are more general and applicable in the machine learning context than metastability results cited in (Şimşekli et al., 2019b); in Section H we see that the regularization effect of truncated heavy-tailed noises described in Theorem 3 is of great generality and can still be observed locally when the irreducibility condition is removed.

## 2.3 $\mathbb{R}^d$ EXTENSIONS OF THE THEORETICAL RESULTS

We focused on the $\mathbb{R}^1$ case so far, for the clarity of the exposition. In this section, we informally reiterate that the same effect under truncated heavy-tailed noises persists in high-dimensions. Rigorous statements are provided in Appendix I. We consider a setting similar to those in Imkeller et al. (2010a) and analyze the first exit time $\sigma(\eta, x) = \min\{k \geq 0 : X_k^\eta(x) \notin \mathcal{G}\}$ from an open, bounded domain $\mathcal{G}$ with smooth boundary. For some $f : \mathbb{R}^d \mapsto \mathbb{R}$, the SGD iterates $X_k^\eta(x) = X_{k-1}^\eta(x) - \varphi_b\big(\nabla f(X_{k-1}^\eta(x)) + \eta Z_k\big)$ are subject to the standard $L_2$ norm clipping with threshold $b > 0$, iid noises $Z_n$ with heavy tails that resemble $1/x^\alpha$ with $\alpha > 1$, and initial condition $X_0^\eta(x) = x$. Assume that the origin $\mathbf{0}$ is the only attractor in $\mathcal{G}$ for the ODE $\dot{\boldsymbol{x}}(t) = -\nabla f(\boldsymbol{x}(t))$. Also, let $l_\mathcal{G}^*$ be the minimum number of jumps required for the ODE $\boldsymbol{x}(t)$ to escape from $\mathcal{G}$ provided that $\boldsymbol{x}(0) = \mathbf{0}$ and the $L_2$ norm of all jumps are less than $b$. The following informal version of Theorem I.2 states that, under the presence of heavy-tailed noises and gradient clipping, the first exit time from a region in $\mathbb{R}^d$ is of order $O(1/\eta^{1+(\alpha-1)l_\mathcal{G}^*})$. Therefore, the order of first exit times from different regions in $\mathbb{R}^d$ are still dictated by the geometric characterization $l_\mathcal{G}^*$, i.e. the minimum number of jumps required for escape, and those with largest $l_\mathcal{G}^*$ may dominate the SGD trajectory as $\eta \downarrow 0$. Furthermore, we extend the analysis to the generalized case where the distribution of noises $Z_n$ exhibits strong preference of certain directions and has different heavy-tailed indices $\alpha$ along different directions. We give the details of the $\mathbb{R}^d$ results in Section I and provide a proof in Section J.

**Theorem 4** (Informal). *Under certain regularity conditions, for Lebesgue almost every $b > 0$, there exist $q > 0$ and $\lambda(\eta)$ that is regularly varying w.r.t $\eta$ with index $1 + (\alpha - 1)l_\mathcal{G}^*$ such that*

$$\lambda(\eta)\sigma(\eta, x) \Rightarrow Exp(q) \text{ as } \eta \downarrow 0 \ \forall x \in \mathcal{G}.$$

## 3 SIMULATION EXPERIMENTS

We empirically demonstrate that, $(a)$ as indicated by Theorem 1, the minimum jump number defined in (3) accurately characterizes the first exit times of the SGDs with clipped heavy-tailed gradient noises; $(b)$ sharp minima can be effectively eliminated from such SGD; and $(c)$ these properties are exclusive to heavy-tails. Under light-tailed noises, SGDs are trapped in sharp minima for a long time. The test function $f \in \mathcal{C}^2(\mathbb{R})$ is the same as in Fig. 1 (Left,e). $m_1$ and $m_3$ are sharp minima in narrow attraction fields, while $m_2$ and $m_4$ are flatter and located in larger attraction fields. Heavy-tailed noises have tail index $\alpha = 1.2$, and *light-tailed* noises are $\mathcal{N}(0, 1)$. See Appendix A for details.

First, we compare the first exit time of heavy-tailed SGD (when initialized at -0.7) from $\Omega_2 = (-1.3, 0.2)$ under 3 different clipping mechanism: (1) $b = 0.28$, where the minimum jump number required to escape is $l^* = 3$; (2) $b = 0.5$, where $l^* = 2$; (3) no gradient clipping, where $l^* = 1$ obviously. According to Theorem 1, the first exit times for the aforementioned 3 clipping mechanism are of order $(1/\eta)^{1.6}$, $(1/\eta)^{1.4}$ and $(1/\eta)^{1.2}$ respectively. These theoretical predictions are accurate as demonstrated in Figure 1 (Right). Next, we investigate the global dynamics of heavy-tailed SGD. We compared the clipped case (with $b = 0.5$) against the case without clipping. Figure 1 (Left, a, b) show the histograms of the empirical distributions of SGD, and Figure 1(Middle, a,b) plots the SGD trajectories. Without gradient clipping, $X_n$ still visits the two sharp minima $m_1, m_3$. Under gradient clipping, the time spent at $m_1, m_3$ is almost completely eliminated and is negligible compared to the time $X_n$ spent at $m_2, m_4$ in larger attraction fields. This matches the predictions of Theorems 2-3: the elimination of sharp minima with truncated heavy-tailed noises. We stress that the said properties are exclusive to heavy-tailed SGD. As shown in Figure 1(Left,c,d) and Figure 1(Middle, c,d), light-tailed SGD are easily trapped at sharp minima for extremely long time.

Figure 3 illustrates the same phenomena in $\mathbb{R}^2$, where $f$ has several saddle points and infinitely many local minima—the local minima of $\Omega_2$ form a line segment, which is an uncountably infinite set. Under clipping threshold $b$, attraction fields $\Omega_1$ and $\Omega_2$ are the *larger* ones since the escape from them requires at least two jumps. This suggests that the theoretical results from Section 2 hold under more general contexts than Assumptions 1-3. In the next section, we provide experimental evidence that suggests that truncated heavy-tailed noise improves the generalization of SGD in deep learning.

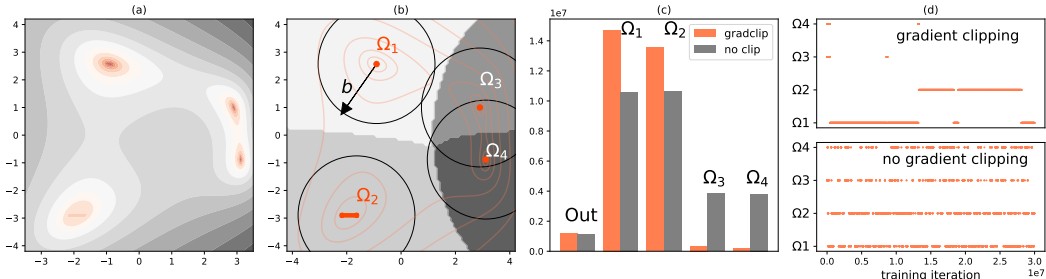

Figure 3: Experiment result of heavy-tailed SGD when optimizing the modified Himmelblau function. **(a)** Contour plot of the test function $f$. **(b)** Different shades of gray are used to indicate the area of the four different attraction fields $\Omega_1, \Omega_2, \Omega_3, \Omega_4$ of $f$. We say that a point belongs to an attraction field $\Omega_i$ if, when initializing at this point, the gradient descent iterates converge to the local minima in $\Omega_i$, which are indicated by the colored dots. The circles are added to imply whether the SGD iterates can escape from each $\Omega_i$ with one large jump or not under clipping threshold $b$. **(c)** The time heavy-tailed SGD spent at different region. An iterate $X_k$ is considered "visiting" $\Omega_i$ if its distance to the local minimizer of $\Omega_i$ is less than 0.5; otherwise we label $X_k$ as "out". **(d)** The transition trajectories of heavy-tailed SGD. The dots represent the last "visited" attraction field at each iteration.

## 4 HEAVY-TAILED SGD IN DEEP LEARNING: AN ABLATION STUDY

In this section, we verify our theoretical results and demonstrate the effectiveness of clipped heavy-tailed noise in training deep neural networks. Contrary to the report in (Şimşekli et al., 2019a), heavy-tailed noise may not be ubiquitous in image classification tasks. For instance, the non-Gaussianity assumption on SGD noise is disputed by experiments in (Panigrahi et al., 2019) for ResNet (see (He et al., 2016)). For tasks considered in this section, the gradient noise is not heavy-tailed when models are randomly initialized (see Appendix A). Motivated by the absence of heavy-tailed noises in image classification, we *make the SGD noise heavy-tailed*. Let $\theta$ be the current model weight during training, $g_{SB}(\theta)$ be the typical small-batch gradient, and $g_{GD}(\theta)$ be the true (deterministic) gradient evaluated on the entire training dataset. Then by evaluating $g_{SB}(\theta) - g_{GD}(\theta)$ we obtain a sample of the gradient noise. Due to the prohibitive cost of evaluating $g_{GD}(\theta)$, we instead use $g_{SB}(\theta) - g_{LB}(\theta)$ as its approximation where $g_{LB}$ denotes the gradient evaluated on a larger batch. This is justified by the unbiasedness in $\mathbb{E}_{LB}[g_{LB}(\theta)] = g_{GD}(\theta)$. For some heavy-tailed random variable $Z$, by multiplying $Z$ with SGD noise, we obtain the following perturbed gradient:

$$g_{heavy}(\theta) \; = \; g_{SB}(\theta) + Z\big(g_{SB*}(\theta) - g_{LB}(\theta)\big) \qquad (6)$$

where $SB$ and $SB*$ are two mini batches that may or may not be identical. We use the following update recursion under gradient clipping threshold $b$: $X_{k+1}^{\eta} = X_k^{\eta} - \varphi_b(\eta g_{heavy}(X_k^{\eta}))$ where $\varphi_b$ is the truncation operator. We consider two different implementations: in *our method 1* (labeled as "our 1" in Table 1), $SB$ and $SB*$ are chosen independently, while in *our method 2* (labeled as "our 2" in Table 1), we use the same batch for $SB$ and $SB*$. In summary, by simply multiplying gradient noise with heavy-tailed random variables, we inject heavy-tailed noise into the optimization procedure.

We conduct an ablation study and benchmark the proposed clipped heavy-tailed methods against the following optimization methods. *LB*: large-batch SGD with $X_{k+1}^{\eta} = X_k^{\eta} - \eta g_{LB}(X_k^{\eta})$; *SB*: small-batch SGD with $X_{k+1}^{\eta} = X_k^{\eta} - \eta g_{SB}(X_k^{\eta})$; *SB + Clip*: the update recursion is $X_{k+1}^{\eta} = X_k^{\eta} - \varphi_b(\eta g_{SB}(X_k^{\eta}))$; *SB + Noise*: Our method 2 WITHOUT the gradient clipping mechanism, leading to the update recursion $X_{k+1}^{\eta} = X_k^{\eta} - \eta g_{heavy}(X_k^{\eta})$.

The experiment setting and choice of hyperparameters are adapted from (Zhu et al., 2019). We consider three different tasks: (1) LeNet (LeCun et al., 1990) on corrupted FashionMNIST (Xiao et al., 2017), (2) VGG11 (Simonyan & Zisserman, 2014) on SVHN (Netzer et al., 2011), (3) VGG11 on CIFAR10 (Krizhevsky et al., 2009) (see Appendix A for details). Here we highlight a few points: First, within the same task, for all the 6 candidate methods will use the same $\eta$, batch size, training iteration, and (when needed) the same clipping threshold $b$ and heavy-tailed RV $Z$ for a fair

Table 1: Test accuracy and expected sharpness of different methods across different tasks. The reported numbers are the averages over 5 replications. For 95% CI, see Appendix A.

| Test accuracy | LB | SB | SB + Clip | SB + Noise | Our 1 | Our 2 |
|---|---|---|---|---|---|---|
| FashionMNIST, LeNet | 68.66% | 69.20% | 68.77% | 64.43% | 69.47% | **70.06%** |
| SVHN, VGG11 | 82.87% | 85.92% | 85.95% | 38.85% | **88.42%** | 88.37% |
| CIFAR10, VGG11 | 69.39% | 74.42% | 74.38% | 40.50% | 75.69% | **75.87%** |
| Expected Sharpness | LB | SB | SB + Clip | SB + Noise | Our 1 | Our 2 |
| FashionMNIST, LeNet | 0.032 | 0.008 | 0.009 | 0.047 | 0.003 | **0.002** |
| SVHN, VGG11 | 0.694 | 0.037 | 0.041 | 0.012 | **0.002** | 0.005 |
| CIFAR10, VGG11 | 2.043 | 0.050 | 0.039 | 2.046 | **0.024** | 0.037 |

Table 2: Our method's gain on test accuracy persists even when applied with techniques such as data augmentation and scheduled learning rates. For 95% CI, see Appendix A.

| CIFAR10-VGG11 | Rep 1 | Rep 2 | Rep 3 | Rep 4 | Rep 5 | Average |
|---|---|---|---|---|---|---|
| SB+Clip | 89.40% | 89.41% | 89.89% | 89.52% | 89.47% | 89.54% |
| Our 1 | 90.76% | 90.57% | 90.49% | 90.85% | 90.79% | **90.67%** |
| Our 2 | 90.67% | 90.23% | 90.52% | 90.13% | 90.70% | 90.45% |
| CIFAR100-VGG16 | Rep 1 | Rep 2 | Rep 3 | Rep 4 | Rep 5 | Average |
| SB+Clip | 55.76% | 56.8% | 56.38% | 56.35% | 56.32% | 56.32% |
| Our 1 | 67.43% | 65.12% | 65.14% | 65.96% | 63.57% | **65.44%** |
| Our 2 | 67.19% | 61.17% | 60.97% | 64.75% | 60.90% | 62.99% |

comparison; the training duration is long enough so that *LB* and *SB* have attained 100% training accuracy and close-to-0 training loss long before the end of training (the exception here is "*SB + Noise*" method; see Appendix A for the details); Second, to facilitate convergence to local minima for *our methods 1 and 2*, we remove heavy-tailed noise for last final 5,000 iterations and run *LB* instead[4].

Table 1 shows that in all 3 tasks both *our method 1* and *our method 2* attain better test accuracy than the other candidate methods. Meanwhile, both methods exhibit similar test performance, implying that the implementation of the heavy-tailed method may not be a the deciding factor. We also report the *expected sharpness* metric $\mathbb{E}_{\nu \sim \mathcal{N}(\mathbf{0}, \delta^2 \mathbf{I})} |L(\theta^* + \nu) - L(\theta^*)|$ used in Zhu et al. (2019); Neyshabur et al. (2017b) where $\mathcal{N}(\mathbf{0}, \delta^2 \mathbf{I})$ is a Gaussian distribution, $\theta^*$ is the trained model weight and $L$ is training loss. In our experiment, we use $\delta = 0.01$ and the expectation is evaluated by averaging over 100 samples. We conduct 5 replications for each experiment scenario and report the averaged performance in Table 1. Smaller sharpness of our methods 1 and 2 confirms that they encourage minimizers with a "flatter" geometry, thus attaining better test performances.

The ablation study in Table 1 shows that both heavy-tailed noise and gradient clipping are necessary to find a flat minima and hence achieve better generalization, which is predicted by our analyses. *SB* and *SB + Clip* achieve similar inferior performances, confirming that clipping does not help when noise is light-tailed. *SB + Noise* injects heavy-tailed noise without gradient clipping, which achieves an inferior performance. This poor performance—even after extensive parameter tuning and engineering (see Appendix A for more details)—demonstrates the difficulty on the optimization front, especially when heavy-tailed noise is present yet little effort is put into controlling the highly volatile gradient noises. This is aligned with the observations in Zhang et al. (2020); Gorbunov et al. (2020) where adaptive gradient clipping methods are proposed to improve convergence of SGD in the presence of heavy-tailed noises. This confirms that gradient clipping is crucial for heavy-tailed SGD.

Lastly, Table 2 shows that even in the more sophisticated settings with training techniques such as data augmentations and scheduled learning rates, truncated heavy-tailed SGD still manages to consistently find solutions with better test performance. For experiment details, see Appendix A. In Table A.5 we also report the sharpness of the obtained solutions.

---

[4]The proposed method can be interpreted as a simplified version of GD + annealed heavy-tailed perturbation, where a detailed annealing is substituted by a two-phase training schedule. In the first *exploration* phase the clipped heavy-tailed noises drive the iterates to explore the loss landscape and identify "wide" attraction fields. In the second *exploitation* phase, removing the artificial perturbation accelerates convergence to local minima.

ACKNOWLEDGEMENT

This work is partially supported by NSF awards DMS-2134012 and CCF-2019844 as a part of NSF Institute for Foundations of Machine Learning (IFML).

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

## A  DETAILS OF NUMERICAL EXPERIMENTS

### A.1  DETAILS OF THE $\mathbb{R}^1$ SIMULATION EXPERIMENT

The function $f$ used in the experiments is

$$f(x) = (x+1.6)(x+1.3)^2(x-0.2)^2(x-0.7)^2(x-1.6)\big(0.05|1.65-x|\big)^{0.6}$$
$$\cdot \Big(1 + \frac{1}{0.01 + 4(x-0.5)^2}\Big)\Big(1 + \frac{1}{0.1 + 4(x+1.5)^2}\Big)\Big(1 - \frac{1}{4}\exp(-5(x+0.8)(x+0.8))\Big).$$

$$(A.1)$$

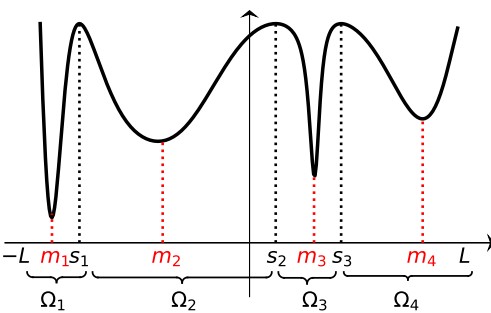

Figure A.1: Illustration of the test function $f$ used in the $\mathbb{R}^1$ experiment.

As shown in Figure A.1, the four isolated local minimizers of $f$ are $m_1 = -1.51, s_1 = -1.3, m_2 = -0.66, s_2 = 0.2, m_3 = 0.49, s_3 = 0.7, m_4 = 1.32$, and in our experiment we restrict the iterates on $[-L, L]$ with $L = 1.6$. The heavy-tailed noises we used in the experiment were $Z_n = 0.1U_nW_n$ where $W_n$ were sampled from Pareto Type II distribution (aka Lomax distribution) with shape parameter $\alpha = 1.2$, and the signs $U_n$ were iid RVs such that $\mathbb{P}(U_n = 1) = \mathbb{P}(U_n = -1) = 1/2$.

In the first exit time experiment, we tested three different settings: (a) $b = 0.28$ (so that $l^* = 3$); (b) $b = 0.5$ (so that $l^* = 2$); (c) no gradient clipping (so that $l_2^* = 1$). For the first case, we tested learning rates $\{0.1, 0.05, 0.03, 0.02, 0.01, 0.005, 0.003, 0.001\}$, while for the other two cases, we tested learning rates $\{0.1, 0.03, 0.01, 0.003, 0.001, 0.0003, 0.0001\}$. For each case, we ran the simulation 20 times and plotted the average of the 20 exit times. Lastly, to prevent excessively long running time of the experiment, the simulation was terminated when the iteration number reached $5 \times 10^7$. This threshold was reached only in the setting with $\eta = 0.001, b = 0.28$.

Next, we present extra sample paths of SGD when applied to function $f$ in eq. (A.1) in Figure A.2 and A.3. The blue curve on the right side of each plot shows $f$ rotated by 90 degrees, and the dashed lines indicate the locations of local minima. For better readability of the figures, we plotted $X_n$ for every 5,000 iterations. To generate these plots, we initialized the SGD iterates at 0.3 (so that it is in $\Omega_3 = (0.2, 0.7)$) and fixed the learning rate as $\eta = 0.001$. Again, we tested both with gradient clipping (with $b = 0.5$) and without gradient clipping. Moreover, we also tested **light-tailed** noises where we use $N(0, 1)$ as the distribution for noises $Z_n$. For each sample path of $X_n$, we run $10,000,000$ iterations. In the left plots of Figure A.2, one can see that with clipped heavy-tailed stochastic gradients, the SGD iterates almost always stay around the wide attraction fields, and the sharp minima are almost completely eliminated from the trajectories of SGD. In comparison, in the right plots of Figure A.2 one can see that without gradient clipping, the heavy-tailed noises will drive SGD to spend substantial amount of time in all the different local minima, including the sharp ones. Lastly, from Figures A.3, one can see that under light-tailed noises and small learning rates, SGD cannot escape a sharp minima once trapped there.

### A.2  DETAILS OF THE $\mathbb{R}^d$ SIMULATION EXPERIMENT

As illustrated in the contour plot in Figure 3 (a), the function $f$ in this experiment is a modified version of Himmelblau function, a commonly used test function for optimization algorithm. The

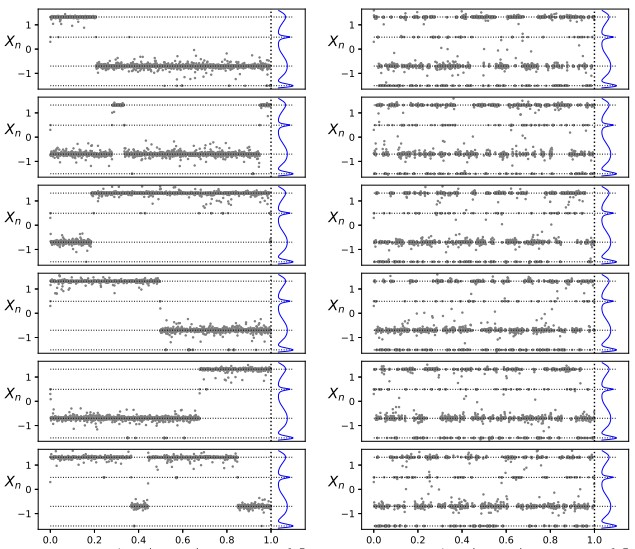

Figure A.2: Five sample paths of SGD under heavy-tailed noises with gradient clipping (left) and without gradient clipping (right). Note that in each case, SGD sample paths exhibit similar patters: SGD almost completely avoided sharp minima with gradient clipping, whereas SGD spent significant amount of time at the sharp minima without gradient clipping.

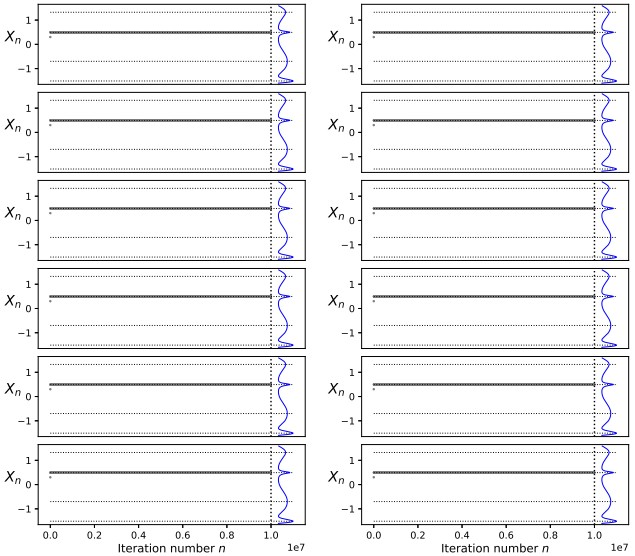

Figure A.3: Five sample paths of SGD under light-tailed noises with gradient clipping (left) and without gradient clipping (right). Note that regardless of the use of gradient clipping, SGD never manages to escape the local minimum that it started from.

modifications serve two purposes. First, as shown in Figure 3 (b), for the modified function the four attraction fields $\Omega_1, \Omega_2, \Omega_3, \Omega_4$ have different sizes; in particular, under gradient clipping threshold $b = 2.15$, from the local minimizers of $\Omega_1$ and $\Omega_2$ (indicated by red dots in the corresponding area) at least two jumps are required to escape from the attraction field, while from the local minimizer in $\Omega_3$ or $\Omega_4$ it is possible to escape with one jump. Therefore, for the minimum jump number required

to escape, we have $l_1^* = l_2^* = 2 > l_3^* = l_4^* = 1$ in this case. Second, for the modified test function $f$, the local minimizer in $\Omega_2$ is not a single point but a connected line segment, which is indicated by the dark line in bottom-left region in Figure 3 (a) and the red line segment in in Figure 3 (b). Therefore, the modification allows us to test the heavy-tailed SGD methods on a more general loss landscape.

Now we describe the construction of the test function $f$. Let $h$ be the Himmelblau function with expression $h(x, y) = (x^2 + y - 11)^2 + (x + y^2 - 7)^2$. Next, define the following transformation for coordinates: $\phi(x, y) = \Big(x(\exp(c_0(x - c_x) + 1)), y(\exp(c_0(x - c_x) + 1))\Big)$. Let the composition be $h_\phi(x, y) = h\Big(\phi(x - a_x, y)\Big)$. To create the connected region of local minimizers, define the following locally "cut" version of $h_\phi$:

$$i(x, y) = \mathbb{1}\{x \in [b_l, b_r], \ |y - a_y| < b_y\},$$
$$h^*(x, y) = (1 - i(x, y))h_\phi(x, y) + i(x, y)\min\{h_\phi(x, y), \ c_1|y - a_y|^{1.1}\}.$$

In other words, by taking minimum of the original $h_\phi$ and a polynomial function w.r.t. $y$ around the original local minimizer of $\Omega_2$, we obtain a function $h^*$ that attains local minimum on an entire line segment with $y = a_y$. Lastly, the test function we use in the experiment is $f = 0.1h^*$, with $a_x = 1.5, a_y = -2.9, b_l = -5.5, b_r = -0.5, b_y = 2.0, c_0 = 0.4, c_1 = 12$.

In the experiment, we initialize the SGD iterates $X_k$ at $X_0 = (2.9, 1.0)$, which is very close to the local minimizer in the small attraction field $\Omega_3$. For both the clipped and unclipped SGD, we perform updates for $3 \times 10^7$ steps, under learning rate $5 \times 10^{-4}$ and heavy-tailed noise $Z_k = 0.75W_k$ where the iid samples $W_k$ are isotropic and the law of $\|W_k\|$, the size of the noise, is Pareto(1.2). For clipped SGD, we use threshold $b = 2.15$. To prevent the iterates from drifting to infinity, after each update $X_k$ is projected back to the $L_2$ ball centered at origin with radius 4.2 whenever $X_k$ leaves this ball.

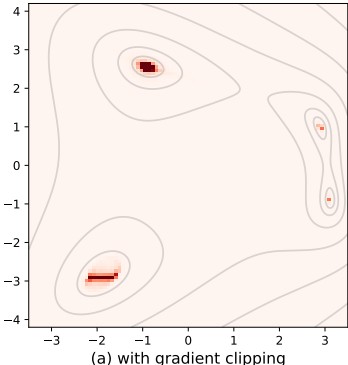
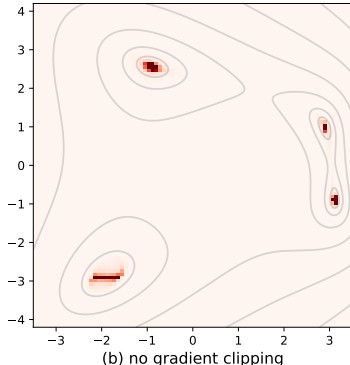

(a) with gradient clipping

(b) no gradient clipping

Figure A.4: Heat map of SGD iterates when optimizing the modified Himmelblau function.

In Figure A.4, we use the $3 \times 10^7$ steps of SGD iterates (for both the clipped and unclipped case) to create heat maps showing locations of SGD iterates. From this figure, two points can be made clear: first, the heavy-tailed SGD does spend much less time at the two small attraction fields when gradient clipping is applied; second, in $\Omega_2$ (the bottom-left attraction field) the SGD iterates frequent the entire connected region of local minima instead of a certain point on this line segment.

## A.3 Details of the ablation study

We first mention that the all experiments using neural networks are conducted on Nvidia GeForce GTX 1080 Ti. For the ablation study, the experiments and scripts are adapted from the ones in Zhu et al. (2019).[5].

In Figure A.5, we display the gradient noise distribution in the three tasks of the ablation study after the model is randomly initialized.

---

[5]https://github.com/uuujf/SGDNoise

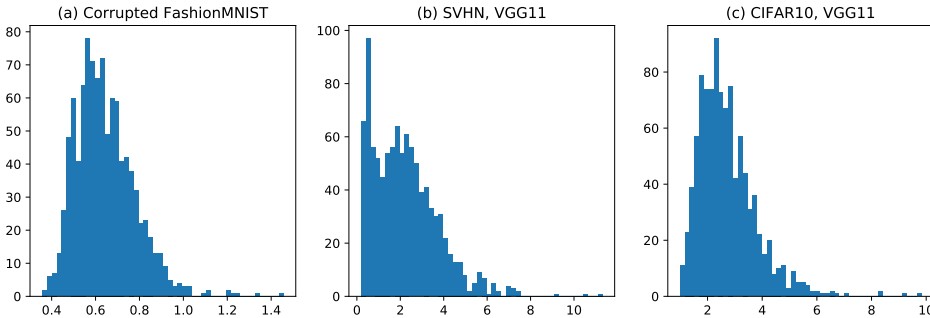

Figure A.5: Distribution of gradient noise in different tasks of the ablation study.

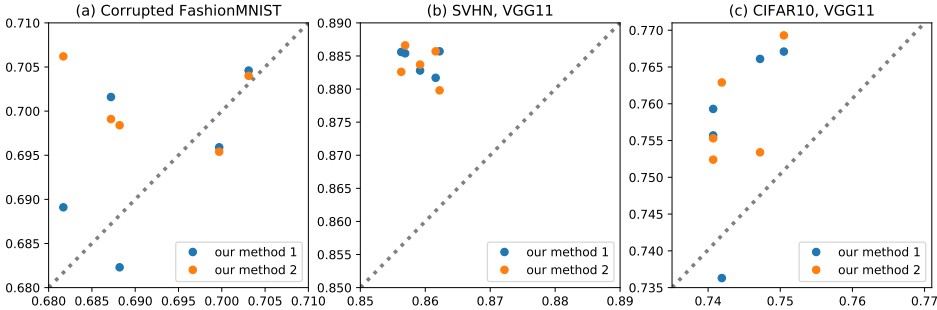

Figure A.6: Test accuracy of the proposed clipped heavy-tailed methods vs. test accuracy of vanilla SGD in the ablation study.

The experiment setting and choice of hyperparameters are mostly adapted from the experiment in Zhu et al. (2019). We consider three different tasks: (1) training LeNet on corrupted FashionMNIST dataset; specifically, we use a 1200-sample subset of the original FashionMNIST training dataset, and for 200 samples points in the training set we randomly assign a label instead of using the correct ones; (2) VGG11 on SVHN dataset, where we use a 25000-sample subset of the training dataset; (3) VGG11 on CIFAR10, where we use the entire training set. For all tasks we use the entire test dataset when evaluating test accuracy.

Table A.1: Test accuracy (percentage) and expected sharpness of different methods across different tasks. The reported numbers are the averages and 95%CI over 5 replications.

| Test Accuracy | Corrupted FMNIST, LeNet | SVHN, VGG11 | CIFAR10, VGG11 |
|---|---|---|---|
| LB | 68.7±0.4 | 82.9±0.4 | 69.4±0.5 |
| SB | 69.2±0.8 | 85.9±0.2 | 74.4±0.4 |
| SB + Clip | 68.8±0.6 | 85.9±0.2 | 74.4±0.8 |
| SB + Noise | 64.4±3.4 | 38.9±24.1 | 40.5±25.1 |
| Our 1 | 69.5±0.8 | **88.4±0.2** | 75.7±1.1 |
| Our 2 | **70.1±0.4** | **88.4±0.2** | **75.9±0.7** |
| Expected Sharpness | Corrupted FMNIST, LeNet | SVHN, VGG11 | CIFAR10, VGG11 |
| LB | 0.032±0.006 | 0.694±0.048 | 2.043±0.083 |
| SB | 0.008±0.001 | 0.037±0.007 | 0.050±0.013 |
| SB + Clip | 0.009±0.001 | 0.041±0.006 | 0.039±0.019 |
| SB + Noise | 0.047±0.02 | 0.012±0.009 | 2.046±2.4 |
| Our 1 | 0.003±0.0003 | **0.002±0.0007** | **0.024±0.005** |
| Our 2 | **0.002±0.0002** | 0.005±0.004 | 0.037±0.007 |

Table A.2: Hyperparameters for training in the ablation study

| Hyperparameters | FashionMNIST, LeNet | SVHN, VGG11 | CIFAR10, VGG11 |
|---|---|---|---|
| learning rate | 0.05 | 0.05 | 0.05 |
| batch size for $g_{SB}$ | 100 | 100 | 100 |
| training iterations | 10,000 | 30,000 | 30,000 |
| gradient clipping threshold | 5 | 20 | 20 |
| $c$ | 0.5 | 0.5 | 0.5 |
| $\alpha$ | 1.4 | 1.4 | 1.4 |

Table A.3: Sharpness of different methods across different tasks. The reported numbers are the averages over 5 replications.

| PAC-Bayes Sharpness | Corrupted FMNIST, LeNet | SVHN, VGG11 | CIFAR10, VGG11 |
|---|---|---|---|
| LB | $5.9 \times 10^3$ | $2.97 \times 10^4$ | $4.87 \times 10^4$ |
| SB | $3 \times 10^3$ | $6.9 \times 10^3$ | $7.2 \times 10^3$ |
| SB + Clip | $3.3 \times 10^3$ | $7.3 \times 10^3$ | $6.8 \times 10^3$ |
| SB + Noise | $3.1 \times 10^3$ | $7.76 \times 10^4$ | $6.74 \times 10^4$ |
| Our 1 | $1.9 \times 10^3$ | $\mathbf{2.1 \times 10^3}$ | $\mathbf{4.8 \times 10^3}$ |
| Our 2 | $\mathbf{1.6 \times 10^3}$ | $2.3 \times 10^3$ | $5.8 \times 10^3$ |
| Maximal Sharpness | Corrupted FMNIST, LeNet | SVHN, VGG11 | CIFAR10, VGG11 |
| LB | $1.01 \times 10^4$ | $3.78 \times 10^4$ | $5.46 \times 10^4$ |
| SB | $4.9 \times 10^3$ | $9.1 \times 10^3$ | $8.5 \times 10^3$ |
| SB + Clip | $5.4 \times 10^3$ | $9.3 \times 10^3$ | $8 \times 10^3$ |
| SB + Noise | $5.4 \times 10^3$ | $1.19 \times 10^5$ | $1.18 \times 10^5$ |
| Our 1 | $3.2 \times 10^3$ | $\mathbf{2.5 \times 10^3}$ | $\mathbf{5.8 \times 10^3}$ |
| Our 2 | $\mathbf{2.5 \times 10^3}$ | $2.8 \times 10^3$ | $6.5 \times 10^3$ |

The heavy-tailed multipliers $Z_n$ used in this experiment, whenever heavy-tailed noise is needed, are $Z_n = cW_n$ where $W_n$ are iid Pareto($\alpha$) RVs. For each task, we first randomly initialize each model, and then run the 6 candidate methods in parallel starting from the same randomly initialized model weights for a fair comparison.

The hyperparameters in training for each task are listed in Table A.2. The same set of hyperparameters is used for all methods in the same task. Whenever gradient clipping scheme is applied, we clip the gradient if its $L_2$ norm exceeds the threshold given in Table A.2. The exception here is the "*SB + Noise*" method: we use learning rate $\eta = 0.005$; for FashionMNIST task we train for 100,000 iterations and the heavy-tailed noise is removed for the final 50,000 iterations; for SVHN and CIFAR10 tasks, we train for 150,000 iterations and heavy-tailed noise is removed for the last 70,000 iterations. Besides, for this method we always clip the model weights if its $L_\infty$ norm exceeds 1. The reason for the extra tuning and extended training in "*SB + Noise*" method is that, without the said modifications, in all three tasks we observed that the model weights quickly drift to infinity and explodes; even with the weight clipping implemented, the model performance stays at random level with no signs of improvements if we do not tune down learning rate.

In Table A.3, we also report the sharpness of solutions under different shaprness metrics. First, the *PAC-Bayes Sharpness* metric (see equation (53) in Jiang et al. (2019)) is defined as $1/\sigma^2$ where $\sigma$ is equal to the smallest $\delta$ that induces a 0.1 expected sharpness, and reflects the sharpness/flatness parameter used in studies on generalization gaps under the PAC-Bayes framework (see Neyshabur et al. (2017a)). Besides, the *Maximal Sharpness* metric (see equation (54) in Jiang et al. (2019)) is defined as $1/\sigma^2$ where $\sigma$ is equal to the smallest radius $\delta$ that makes $\max_{\|\nu\|_\infty \leq \delta} |L(\theta^* + \nu) - L(\theta^*)| \geq 0.1$, and metrics of form $\max_{\|\nu\| \leq \delta} |L(\theta^* + \nu) - L(\theta^*)|$ can be considered as a proxy for the spectral norm of the Hessian at the solution (see Dinh et al. (2017)). It worth noticing that, for all three sharpness metrics, the smaller the value is the "flatter" the loss landscape is around the solution. Lastly, for evaluation of the PAC-Bayes Sharpness and Maximal Sharpness metrics, we conduct binary search as in Algorithm 2 of Jiang et al. (2019) with $\epsilon_d = 0.01, \epsilon_\sigma = 0, M_1 = 10$ and $M_2 = 100$; in our setting we always evaluate the training loss using one sweep of the entire training set, so $M_3$ is a

Table A.4: Results and 95% CI in the experiments with data augmentation.

| Test Accuracy | SB + Clip | Our 1 | Our 2 |
|---|---|---|---|
| CIFAR10, VGG11 | 89.5±0.2 | **90.7±0.1** | 90.5±0.2 |
| CIFAR100, VGG16 | 56.3±0.3 | **65.4±1.2** | 63.0±2.5 |
| Expected Sharpness | SB + Clip | Our 1 | Our 2 |
| CIFAR10, VGG11 | 0.17±0.005 | **0.09±0.004** | 0.10±0.003 |
| CIFAR100, VGG16 | 0.86±0.02 | **0.44±0.05** | 0.48±0.07 |

Table A.5: Sharpness of solutions obtained by different methods in CIFAR10/100 tasks with data augmentation. Numbers reported here are the average of 5 replications.

| CIFAR10-VGG11 | SB + Clip | Our 1 | Our 2 |
|---|---|---|---|
| Expected Sharpness | 0.167 | **0.085** | 0.096 |
| PAC-Bayes Sharpness | $1.31 \times 10^4$ | $\mathbf{9 \times 10^3}$ | $10^4$ |
| Maximal Sharpness | $1.66 \times 10^4$ | $1.29 \times 10^4$ | $\mathbf{1.22 \times 10^4}$ |
| CIFAR100-VGG16 | SB + Clip | Our 1 | Our 2 |
| Expected Sharpness | 0.857 | **0.441** | 0.479 |
| PAC-Bayes Sharpness | $2.49 \times 10^4$ | $\mathbf{1.9 \times 10^4}$ | $1.98 \times 10^4$ |
| Maximal Sharpness | $2.75 \times 10^4$ | $\mathbf{2.12 \times 10^4}$ | $2.16 \times 10^4$ |

case-specific and is equal to the number of batches of the training set under the batch size for the task at hand.

In Figure A.6, we plot the test accuracy of our method against that of the SGD for all 5 replications and 3 tasks.

## A.4 DETAILS OF CIFAR10/100 EXPERIMENTS WITH DATA AUGMENTATION

For both methods, we train the model for 300 epochs and set the initial learning rate as 0.1. In our method, the training can be partitioned into two phases. In the first phase (the first 200 epochs), the learning rate is kept at a constant. In the second phase, for every 30 epoch we reduce the learning rate by half. Also, an $L_2$ weight decaying with coefficient $5 \times 10^{-4}$ is enforced. As for parameters for heavy-tailed noises in eq. (6), we use $c = 0.5$ and $\alpha = 1.4$ in the first phase, and in the second phase we remove heavy-tailed noise and use *SB* to update weights. In both methods for the small-batch direction $g_{SB}$ the batch size is 128, while for $g_{LB}$ we evaluate the gradient on a large sample batch of size 1,024. Under the epoch number 300 and batch size 128, the count of total iterations performed during training is $1.17 \times 10^5$. To augment the dataset, random horizontal flipping and cropping with padding size 4 is applied for each training batch. Lastly, gradient clipping scheme is applied for both methods, and we fix $b = 0.5$. In other words, when the learning rate is $\eta$ (note that due to the scheduling of learning rates, $\eta$ will be changing throughout the training), the gradient is clipped if its $L_2$ norm is larger than $b/\eta$. The scripts are adapted from the ones in https://github.com/chengyangfu/pytorch-vgg-cifar10.

These results are presented in Table 2. Furthermore, in Table A.5 we see that our truncated heavy-tailed method also manages to find solutions with a flatter geometry.

## A.5 DISCUSSION ON GRADIENT NOISE DISTRIBUTIONS IN THE EXPERIMENTS

In this subsection, we (i) present the empirical evidence that supports our characterization of the baseline model—i.e., the absence of heavy tails in the gradient noise—and (ii) clarify that the emergence of heavy-tails in the *stationary distribution* of SGD (argued in Hodgkinson & Mahoney (2020); Gurbuzbalaban et al. (2020)) does not contradict the observed absence of heavy tails in the *gradient noise* of SGD. This allows us to study the impact of truncated heavy-tails in the gradient noise separately from the choice of hyper-parameters.

We start with our observations on the noise distributions. In view of the well-established wisdom in heavy-tail literature that there is no single perfect tail estimator, we analyzed the stochastic gradient noise with four different methods: QQ plot, empirical mean residual life (EMRL), Hill plot, and

PLFIT (Clauset et al., 2009). We applied these estimation/diagnostic tools (i) at the beginning of the training, (ii) halfway through the training, (iii) at the end of the training. Throughout all our experiments, we consistently observe strong evidence that the gradient noises are light-tailed, and even if (against all odds) the noises were from a heavy-tailed distribution, the tail index should be far greater (hence, the resulting tail is much lighter) than the heavy-tails we inject (or the popular alpha stable assumption), and hence, the point we make with our tail-inflation experiment is still valid. We summarize the results as follows.

First, QQ plots below (Fig. A.7-A.21) clearly show that the tails in noise distribution are always much lighter than the Pareto distributions with $\alpha = 2$ or even 10. Therefore, the typical power-law assumption, especially the alpha-stable distributions in Şimşekli et al. (2019b) (with $\alpha \in (0, 2)$), seems far from the distribution of the actual data we obtained in the image classification tasks. In fact, the tail of the noise distributions seems to be between that of lognormal and normal distributions, implying that it is lighter than any power-law distribution.

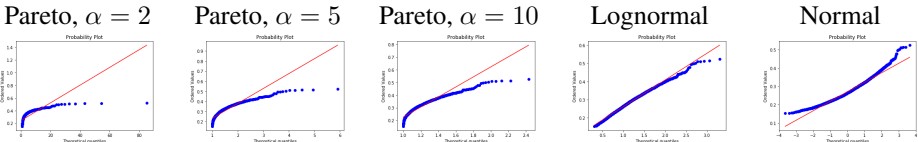

Figure A.7: Ablation Study, Corrupted FMNIST & LeNet: At the beginning

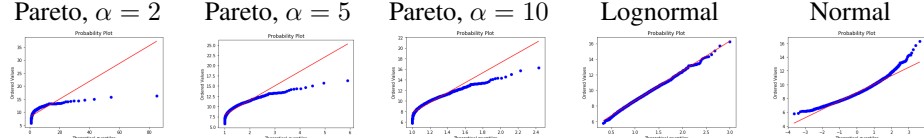

Figure A.8: Ablation Study, Corrupted FMNIST & LeNet: Half way through the training

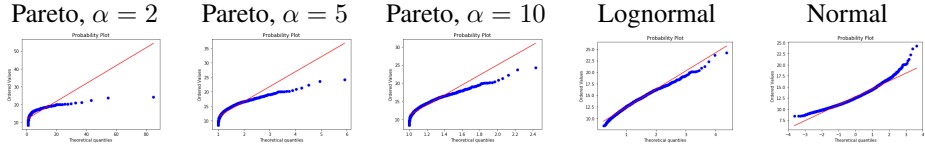

Figure A.9: Ablation Study, Corrupted FMNIST & LeNet: At the end of training

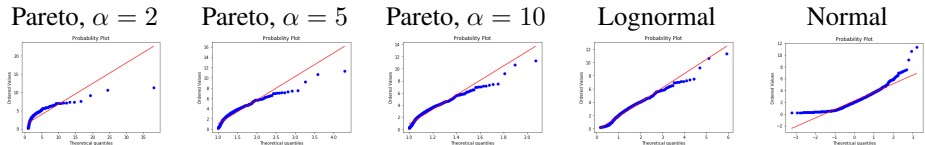

Figure A.10: Ablation Study, SVHN & VGG11: At the beginning

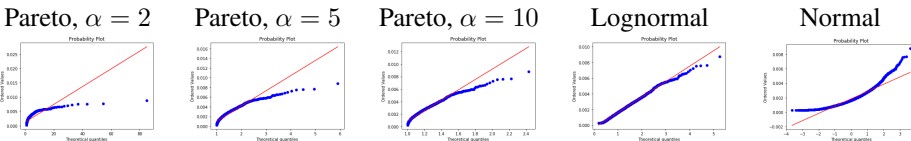

Figure A.11: Ablation Study, SVHN & VGG11: Half way through the training

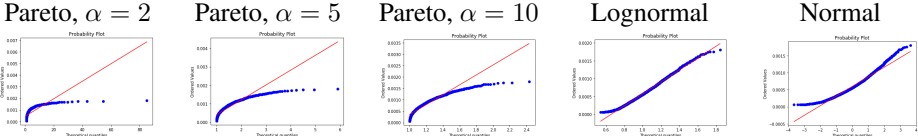

Figure A.12: Ablation Study, SVHN & VGG11: At the end of training

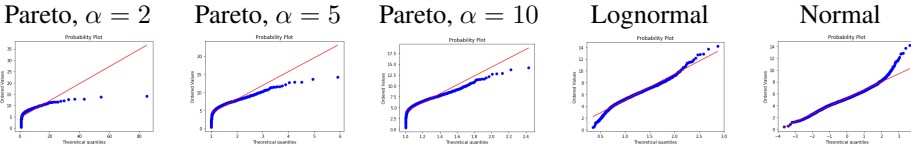

Figure A.13: Ablation Study, CIFAR10 & VGG11: At the beginning

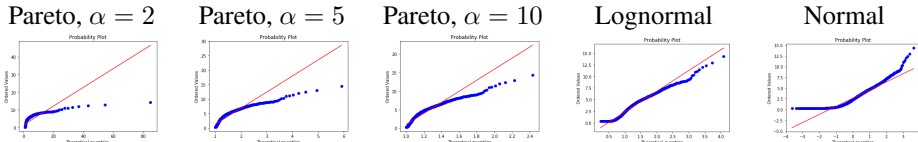

Figure A.14: Ablation Study, CIFAR10 & VGG11: Half way through the training

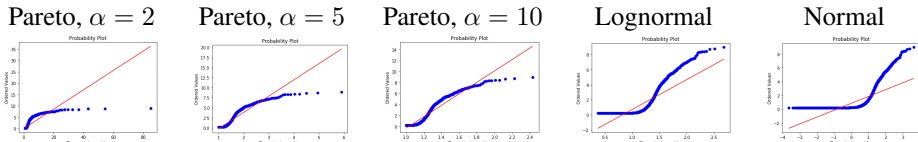

Figure A.15: Ablation Study, CIFAR10 & VGG11: At the end of training

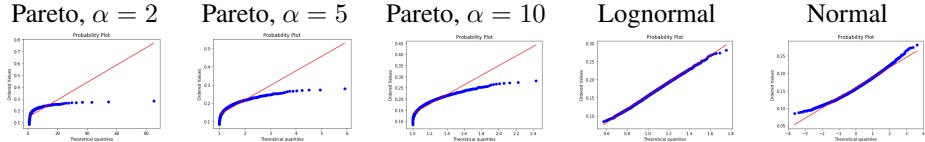

Figure A.16: Data Augmentation, CIFAR10 & VGG11: At the beginning

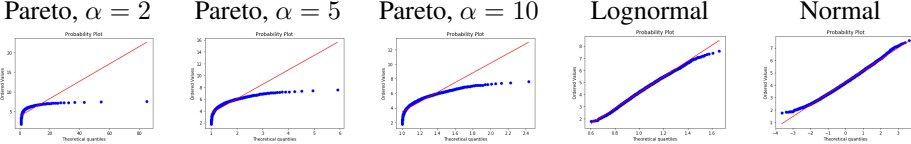

Figure A.17: Data Augmentation, CIFAR10 & VGG11: Half way through the training

Next, we plotted the empirical mean residual life (EMRL) of the gradient noise distributions in Fig. A.22-A.26. It is well known that the mean residual life blows up to infinity if and only if the distribution is heavy-tailed (more precisely, long-tailed). However, from the figures, one can see that none of the EMRL exhibits such a pattern in any case tested in our experiments. Instead, we see clear downward trends, which strongly suggest light tails, in all the tested cases.

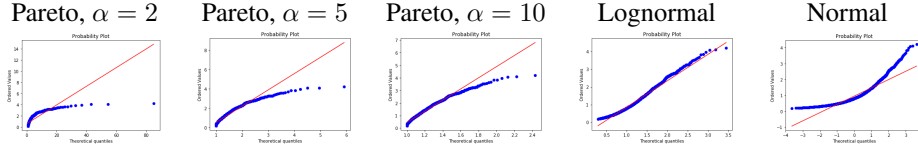

Figure A.18: Data Augmentation, CIFAR10 & VGG11: At the end of training

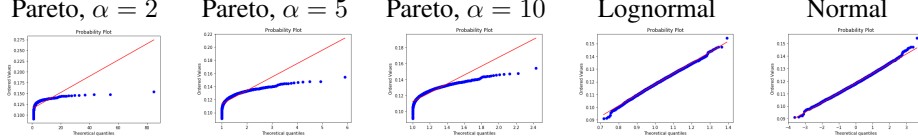

Figure A.19: Data Augmentation, CIFAR100 & VGG16: At the beginning

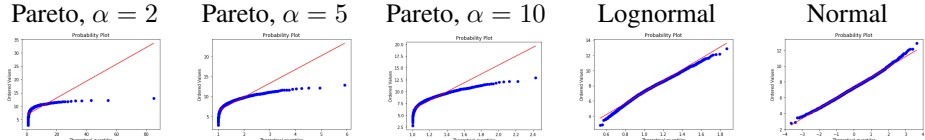

Figure A.20: Data Augmentation, CIFAR100 & VGG16: Halfway through the training

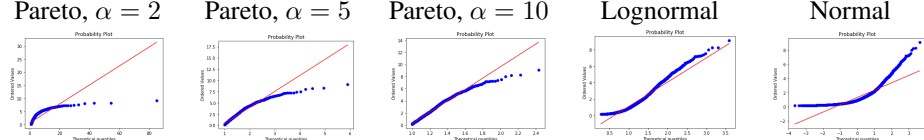

Figure A.21: Data Augmentation, CIFAR100 & VGG16: At the end of training

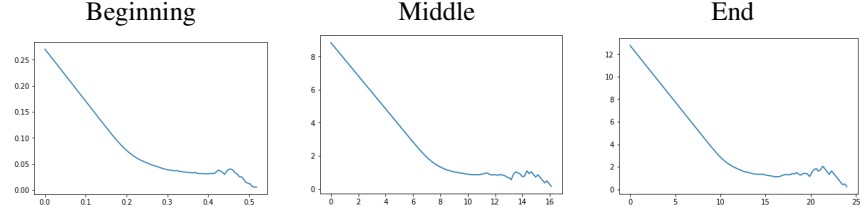

Figure A.22: Plots of empirical mean residual life for noises in FMNIST&LeNet Task throughout training

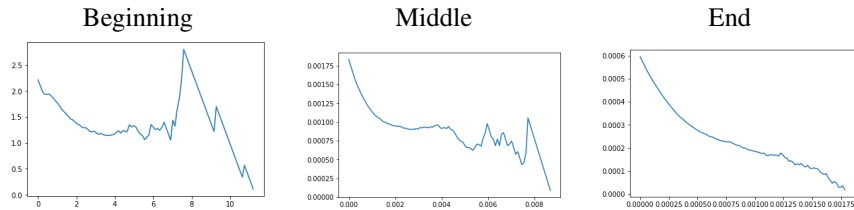

Figure A.23: Plots of empirical mean residual life for noises in SVHN&VGG 11 Task throughout training

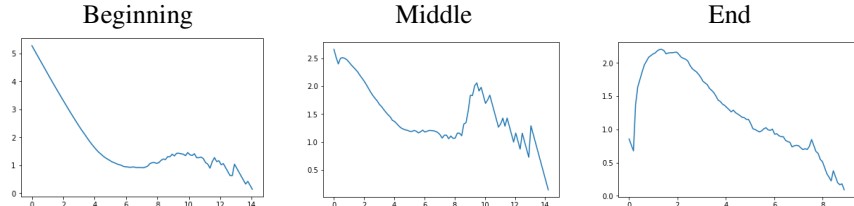

Figure A.24: Plots of empirical mean residual life for noises in CIFAR 10&VGG 11 Task throughout training

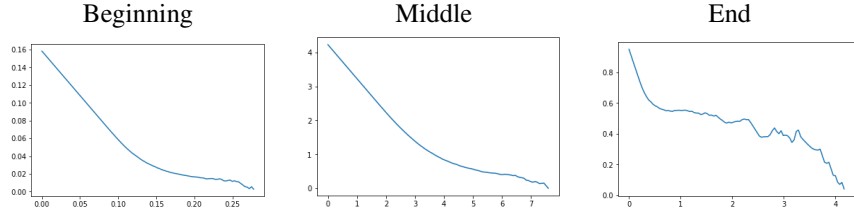

Figure A.25: Plots of empirical mean residual life for noises in dataAug, CIFAR 10&VGG 11 Task throughout training

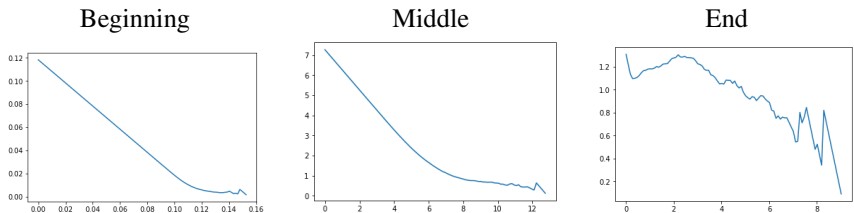

Figure A.26: Plots of empirical mean residual life for noises in dataAug, CIFAR 100&VGG 11 Task throughout training

If we assume a power-law tail, the Hill estimator is a classical tail index estimator with a long history in extreme value theory literature. A critical algorithmic parameter of the Hill estimator is the number of order statistics used in the estimation, and the Hill plot is a popular exploratory tool for investigating the Hill estimators with different numbers of order statistics. Although it is well known that Hill estimators and Hill plots are fallible if the power-law assumption is not satisfied, (and hence, it is not suited for deciding whether a given set of samples are from light-tailed distribution or heavy-tailed distribution; in particular, the method will return some power-law tail index $\alpha$ whether or not the samples are from a heavy-tailed distribution or a light-tailed one) we present the Hill plot to see what would be the estimated tail indices if the gradient noises hypothetically followed a power law. In the Hill plots shown in Fig. A.27-A.31, we presented the rescaled version (altHill) of the Hill plots (see Chapter 4.4 in (Resnick, 2007)) for the following reason. Hill estimator is a consistent estimator of the power-law index when the proportion of the samples used approaches 0, and altHill plots allow us to scrutinize the estimated indices under a small proportion of samples. In particular, the points around the red dashed lines correspond to estimation using the top 1% of the samples. We can see that most Hill plots stay well above 10 for the most part and almost never drop below 2. This strongly suggests that even if the gradient noises are from a heavy-tailed distribution, it is likely to have a very high power-law index (implying relatively lighter tails), and hence, we cannot expect to observe a prominent heavy-tailed behavior from them.

A popular data-driven approach with statistical guarantees (again, under the assumption that the samples are indeed from a heavy-tailed distribution) to select the number of order statistics in the Hill plot is PLFIT (Clauset et al., 2009). We estimated the power-law indices using the python implementation (Alstott et al., 2014) of PLFIT. The numbers are presented in Table A.6. All the

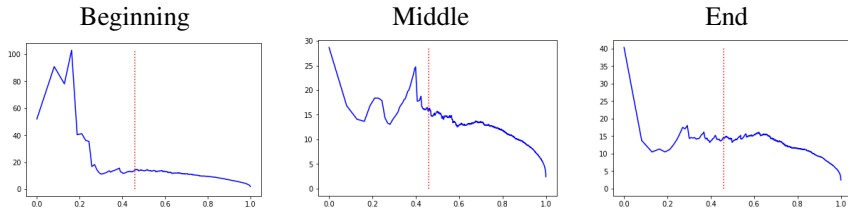

Figure A.27: altHill Plots for noises in FMNIST&LeNet Task throughout training. Dashed Red Line: Estimation based on the largest $1\%$ data

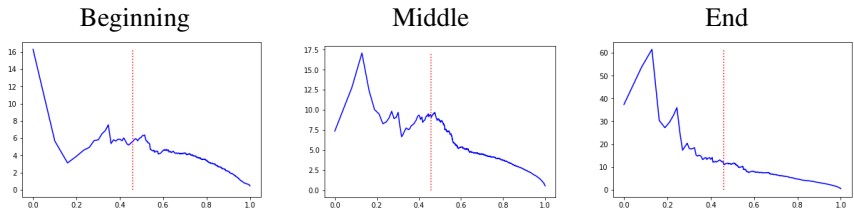

Figure A.28: altHill Plots for noises in SVHN&VGG 11 Task throughout training. Dashed Red Line: Estimation based on the largest $1\%$ data

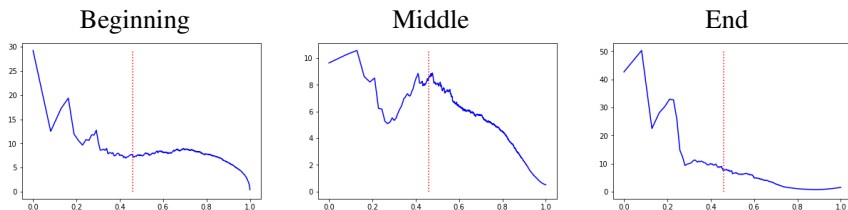

Figure A.29: altHill Plots for noises in CIFAR 10&VGG 11 Task throughout training. Dashed Red Line: Estimation based on the largest $1\%$ data

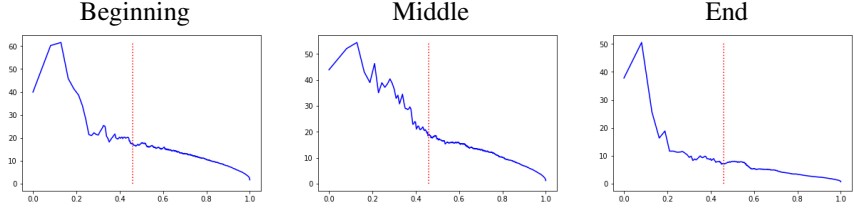

Figure A.30: altHill Plots for noises in Data Augmentation, CIFAR 10&VGG 11 Task throughout training. Dashed Red Line: Estimation based on the largest $1\%$ data

estimations are at least 5 for all cases tested in our experiments, and most of the time, the estimation is above 10. Again, this means that even under the hypothetical assumption (against what the QQ plots and EMRLs suggest) that the gradient noises were from a heavy-tailed distribution, the tail indices of the gradient noises should be large, and hence, the gradient noises in our experiments have much lighter tails than any $\alpha$-stable distribution (which requires $\alpha < 2$) or the heavy-tailed noises we injected during tail inflation experiments ($\alpha = 1.4$). In summary, extensive statistical analyses above suggest the absence of heavy tails in the gradient noises in our experiments. Therefore, the characterization of the vanilla SGD as the light-tailed (or at least lighter than the inflated tails) benchmark in our experiments is valid, and our ablation study is well grounded.

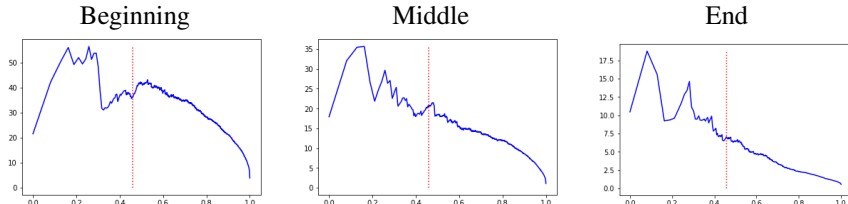

Figure A.31: altHill Plots for noises in Data Augmentation, CIFAR 100&VGG 16 Task throughout training. Dashed Red Line: Estimation based on the largest $1\%$ data

Table A.6: Power-law Indices Estimation throughout the Training, using PLFIT. All the estimations are at least 5 for all cases tested in our experiments, and most of the times the estimation is above 10. This means that even under the assumption that the gradient noises were from a heavy-tailed distribution, they should have much lighter tails than any $\alpha$-stable distribution (which requires $\alpha < 2$) or the heavy-tailed noises we injected during tail inflation experiments ($\alpha = 1.4$).

| Task | Beginning | Middle | End |
|---|---|---|---|
| FMNIST, LeNet | 14.3 | 14.2 | 16.5 |
| SVHN, VGG11 | 5.0 | 5.2 | 12.5 |
| CIFAR10, VGG11 | 9.2 | 6.6 | 7.0 |
| dataAug, CIFAR10, VGG11 | 16.2 | 16.2 | 8.5 |
| dataAug, CIFAR100, VGG16 | 35.1 | 14.4 | 5.35 |

Next, we compare our work to recent literature on heavy-tailed phenomena in stationary distribution of SGD; see, for instance, Hodgkinson & Mahoney (2020); Gurbuzbalaban et al. (2020). To be specific, Hodgkinson & Mahoney (2020); Gurbuzbalaban et al. (2020) show that heavy tails can arise in the stationary distribution of SGD through multiplicative dynamics, and the tail index of the resulting stationary distribution can be characterized by the learning rates and the magnitude of noises (through batch size). However, the results in Hodgkinson & Mahoney (2020); Gurbuzbalaban et al. (2020) do not imply the existence of heavy tails in the gradient noises. In both papers (as well as other works in the literature), the "heaviness" of the tail of the gradient noise (e.g., power-law index $\alpha$ in heavy-tailed cases) is fixed in the model (same as in our setting) and not entangled with the learning rate or batch size. For example, $B_k$ in (5) of Hodgkinson & Mahoney (2020) corresponds to the gradient noise, and its tail index doesn't depend on the learning rate or the batch size. In particular, the change of learning rate does not induce heavy tails in $B_k$. On the other hand, we focus on the impact of the heavy-tails *in the gradient noise* and the truncation scheme for (any) fixed batch size and small learning rates on the global dynamics of SGD. In view of this, it should be clear that our analysis can be decoupled from the choice of batch size or the impact of the learning rate on SGD's stationary distribution. Therefore, our observation and the design of the experiments are compatible with the aforementioned references.

## B  IMPLICATIONS OF THE THEORETICAL RESULTS

**Systematic control of the exit times from attraction fields:** In light of the wide minima folklore, one may want to find techniques to modify the sojourn time of SGD at each attraction field. Theorem 1 suggests that the order of the first exit time (w.r.t. learning rate $\eta$) is directly controlled by the gradient clipping threshold $b$. Recall that for an attraction field with minimum jump number $l^*$, Theorem 1 tells us the exit time from this attraction field is roughly of order $(1/\eta)^{1+(\alpha-1)l^*}$. Given the width of the attraction field, its minimum jump number $l^*$ is dictated by gradient clipping threshold $b$. Therefore, gradient clipping provides us with a very systematic method to control the exit time of each attraction field. For instance, given clipping threshold $b$, the exit time from an attraction field with width less than $b$ is of order $(1/\eta)^{\alpha}$, while the exit time from one larger than $b$ is at least $(1/\eta)^{2\alpha-1}$, which dominates the exit time from smaller ones.

**The role of structural properties of $\mathcal{G}$ and $f$:** Recall that in order for Theorem 3 to apply, the irreducibility of $\mathcal{G}$ is required. Along with the choice of $b$, the geometry of function $f$ is a deciding factor of the irreducibility. For instance, we say that $\mathcal{G}$ is *symmetric* if for any attraction field $\Omega_i$ such that $i = 2, 3, \cdots, n_{\min} - 1$ (so that $\Omega_i$ is not the leftmost or rightmost one at the boundary), we have $q_{i,i-1} > 0, q_{i,i+1} > 0$. One can see that $\mathcal{G}$ is symmetric if and only if, for any $i = 2, 3, \cdots, n_{\min} - 1$, $|s_i - m_i| \vee |m_i - s_{i-1}| < l_i^* b$, and symmetry is a sufficient condition for the irreducibility of $\mathcal{G}$. The graph illustrated in Figure 2(Middle) is symmetric, while the one in Figure 2(Right) is not. As the name suggests, in the $\mathbb{R}^1$ case the symmetry of $\mathcal{G}$ is more likely to hold if the shape of attraction fields in $f$ is also nearly symmetric around its local minimum. If not, the symmetry (as well as irreducibility) of $\mathcal{G}$ can be violated as illustrated in Figure 2, especially when a small gradient clipping threshold $b$ is used.

Generally speaking, our results imply that, even with the truncated heavy-tailed noises, the function $f$ needs to satisfy certain regularity conditions to ensure that SGD iterates avoid undesirable minima. This is consistent with the observations in Li et al. (2018b): the deep neural nets that are more trainable with SGD tend to have a much more regular structure in terms of the number and shape of local minima.

**Heavy-tailed SGD without gradient clipping:** It is worth mentioning that our results also characterize the dynamics of heavy-tailed SGDs without gradient clipping. For instance, since the reflection operation at $\pm L$ restricts the iterates on the compact set $[-L, L]$, if we use a truncation threshold $b$ that is large than $2L$, then any SGD update that moves larger than $b$ will definitely be reflected at $\pm L$. Therefore, the dynamics are identical to that of the following iterates without gradient clipping:

$$X_n^{\eta,\text{unclipped}} = \varphi_L\left(X_{n-1}^{\eta,\text{unclipped}} - \eta f'(X_{n-1}^{\eta,\text{unclipped}}) + \eta Z_n\right). \tag{B.1}$$

The next result follows immediately from Theorem 1 and H.2.

**Corollary B.1.** *There exist constants $q_i > 0 \; \forall i$, $q_{i,j} > 0 \; \forall j \neq i$ such that the following claims hold for any $i$ and any $x \in \Omega_i$.*

1) *Under $\mathbb{P}_x$, $q_i H(1/\eta)\sigma_i(\eta)$ converges in distribution to an Exponential random variable with rate 1 as $\eta \downarrow 0$;*

2) *For any $j = 1, 2, \cdots, n_{min}$ such that $j \neq i$,*
$$\lim_{\eta \downarrow 0} \mathbb{P}_x(X_{\sigma_i(\eta)}^\eta \in \Omega_j) = q_{i,j}/q_i.$$

3) *Let $Y$ be a continuous-time Markov chain on $\{m_1, \cdots, m_{n_{min}}\}$ with generator matrix $Q$ parametrized by $Q_{i,i} = -q_i, Q_{i,j} = q_{i,j}$. Then*
$$X_{\lfloor t/H(1/\eta) \rfloor}^{\eta,\text{unclipped}}(x) \to Y_t(m_i) \quad \text{as } \eta \downarrow 0$$
*in the sense of finite-dimensional distributions.*

At first glance, Corollary B.1 may seem similar to the results in Şimşekli et al. (2019a) and Pavlyukevich (2007). However, the object studied in Şimşekli et al. (2019a); Pavlyukevich (2007) is different: they study the following Langevin-type stochastic differential equation (SDE) driven by an $\alpha$-stable Lévy process $L_t$ with scaling factor $\eta > 0$:
$$dY_t^\eta = -f'(Y_{t-}^\eta)dt + \eta dL_t.$$

In particular, Pavlyukevich (2007) studies the metastability of $Y_t^\eta$ and concludes that as $\eta \downarrow 0$, the first exit time and global dynamics of $Y_t^\eta$ admit a similar characterization as described in our Theorem B.1. Then Theorem 4 in Şimşekli et al. (2019a) argues that when the learning rate $\eta$ is sufficiently small, the distribution of the first exit time of the SGD $X_n^\eta$ and that of the Lévy-driven Langevin SDE $Y_t^\eta$ are similar. However, the analysis of Şimşekli et al. (2019a) hinges critically on the assumption that $L_t^\alpha$ is symmetric and $\alpha$-stable. While such an assumption is convenient for their analysis, it is a strong assumption. It implies that the gradient noise distribution belongs to a very specific parametric family and excludes all the other heavy-tailed distributions. In particular, the assumption precludes analysis of any heavy tails with finite variance. On the contrary, our work directly analyzes the SGD $X_n^\eta$ and reveals the heavy-tailed SGD dynamics at a much greater level of generality. Specifically, we allow the noise to have general regularly varying distributions with arbitrary tail index—which includes $\alpha$-stable distributions as a (very) special case—and extend the characterization of global dynamics of heavy-tailed SGD to the adaptive versions of SGD where gradient clipping is applied.

## C  Geometric Characterization $l^*$ and Existing Sharpness Metrics

Throughout the paper, we have been using the term *sharp minima* when describing the elimination effect of truncated heavy-tailed SGD, while appealing to $l^*$, the minimum number of jumps requires for escape defined in eq. (3), when rigorously presenting our theoretical results about dynamics of truncated heavy-tailed SGD. Despite the possible ambiguity of its use in the main paper, the terminology *sharpness* is meant to familiarize our rather technically involved notion of how heavy-tailed SGD behave. To resolve the potential confusions, in this section we provide a detailed discussion on the relationship between our geometric characterization $l^*$ and the existing sharpness metrics.

In light of recent discovery that all local minima might be global minima for over-parametrized deep neural nets (see Li et al. (2018a)), one popular explanation for the generalization gap between different local minima (for instance, between solutions found by GD and SGD) is that the geometry around the solution is closely related to its performance in the test setting. After the seminal work by Keskar et al. (2016), a myriad of attempts have been made to provide empirical or theoretical evidences for the link between the sharpness of a local minimum and its generalization performance; See, for instance, Xie et al. (2020); Jiang et al. (2019); He et al. (2019); Zhou et al. (2020). In summary, there exist at least four different classes of sharpness metrics among the current literature:

(a) spectral norm of the Hessian at the local minimum, one surrogates of which is the *maximal sharpness* type of metrics; a variant used in Xie et al. (2020) based on corresponding large deviation theory is the eigenvalue of the Hessian along certain escape directions;

(b) *expected sharpness* type of metrics that evaluate the general fluctuation of the loss function within a $L_p$ ball of centered at the local minimum (see Zhu et al. (2019));

(c) complexity metrics based on PAC-Bayes theory (see, for example, Neyshabur et al. (2017a));

(d) geometric property of a domain or the entire attraction field instead of the local minimum itself; see the mass of a Radon measure $m(W)$ over the domain $W$ in Theorem 1 in Zhou et al. (2020).

Under such taxonomy, the $l^*$ characterization proposed in this paper falls into category (d). Indeed, it describes the minimum effort required for escaping the domain rather than the geometry merely around a certain neighborhood of the local minimum. Here we have two remarks on the definition of $l^*$. First, for a given loss landscape, this quantity $l^*$ is dictated by the gradient clipping threshold $b > 0$. In fact, this concept is tailored for the gradient clipping case, as in the unclipped case with heavy-tailed SGD, for any attraction field $\Omega_i$ we always have $l_i^* = 1$. Second, it is analogous to the term $m(W)$ in Zhou et al. (2020) in the sense that it reflects the *volume* of the attraction field (when compared to a given threshold $b$). Therefore, we stress that a more precise interpretation of $l^*$, as well as other metrics in class (d), is that they characterize how *wide* or *narrow* each minimum (attraction field) is, and a more a clear description of our theoretical results Theorem 1-3 is that truncated heavy-tailed SGD effectively avoids all the *narrow* minima when learning rate is small enough.

While one can intuitively see that wide minima are more likely to be flat ones, we acknowledge the existence of counterexamples where a sharp minimum lies in a wide attraction field. Nevertheless, as the variety of definitions for sharpness keeps growing in existing literature, it becomes rather unlikely for one geometric characterization to always agree with (or be equivalent to) other existing approaches. More importantly, the aim of this paper is not to establish $l^*$ as the orthodox geometric property when studying generalization gap. Instead, as demonstrated in Table 1, A.3 and A.5, in modern deep learning tasks the truncated heavy-tailed method (which prefers solutions with high $l^*$ as indicated by our theoretical results) obtains solutions that generalize better and exhibit *flatter* geometry when evaluated under different sharpness metrics. This observation is well aligned with the recent large-scale empirical study in Jiang et al. (2019), suggesting that in typical training setting it is beneficial to find wider minima (characterized by $l^*$) in order to achieve a flatter geometry and better generalization performance.

## D    On the Relationship Between Sharpness and Generalization

In principle, a sharp minimum does not necessarily lead to poor generalization in the sense that it is possible to construct pathological counterexamples in theory or in specific experiment settings; see, for example, Dinh et al. (2017); Neyshabur et al. (2017a). However, the loss geometry that arises in practice seems to exhibit a strong correlation between the sharpness and the test error. For instance, sharpness-aware optimization methods (Foret et al., 2020; Kwon et al., 2021) improve generalization across various tasks and achieve state-of-the-art performance on the CIFAR dataset. Also, one of the authors of the aforementioned paper Neyshabur et al. (2017a) continued investigating different complexity measures and conducted much larger scale experiments in Jiang et al. (2019). Among more than 40 complexity measures from theoretical and empirical studies in the literature tested in Jiang et al. (2019), the sharpness metrics are the top ones for predicting the generalization performance. In our own experiments, we also show that sharpness and test accuracy are highly correlated.

## E    Stability-driven Analyses on SGD

Wu et al. (2018) takes the perspective of linear stability and establishes conditions for SGD to be attracted to or avoid certain solutions based on learning rate, batch size, and the concept of non-uniformity of local minima. Inspired by (Wu et al., 2018), (Jastrzebski et al., 2020) analyzes the trajectory-wise stability of SGD and found that a "break-even point" partitions the training procedure into two phases: the implicit regularization effects in SGD due to a larger learning rate becomes visible in the second phase after this "break-even point". Similarly, (Cohen et al., 2021) reports that typical GD trajectories in standard image classification tasks can be partitioned into two phases: in the first "progressive sharpening" phase the sharpness of the Hessian monotonically increases, while in the second phase we observe the "edge of stability" regime where the sharpness of Hessian hovers slightly above the critical value $2/\eta$ and the training loss slowly decrease in an oscillating fashion.

Compared to the aforementioned stability analyses, one major difference of our heavy-tailed regime is that, even under gradient clipping, the trajectory cannot be partitioned into such "phases" and the traditional sense of stability around certain local minima is nullified by the constant basin hopping and exploration behaviors under heavy-tailed noises. In particular, the polynomial order of the exit time in heavy-tailed SGD dictates that, when compared to exponentially long exit time in vanilla SGD (see Xie et al. (2020)), the iterates won't become "stabilized" and keep staying around a certain region in a typical training procedure. Instead, our work characterizes the metastability of the truncated heavy-tailed SGD (i.e. constantly transitioning between different wide minima) that cannot be observed in GD or light-tailed SGD. More importantly, compared to the aforementioned stability related analyses, our work provides a very tight characterization of the global dynamics and distributions of the entire sample path.

## F    Existing Analyses on Heavy-tailed Phenomena in SGD

We start with one clarification: what has been established in Hodgkinson & Mahoney (2020); Gurbuzbalaban et al. (2021); Wojtowytsch (2021); Mori et al. (2021) is that the stationary distribution of the SGD (or the corresponding continuous-time SDE) can be heavy-tailed, and the heavy tails therein can depend on various training hyperparameters. This, however, does not imply that the gradient noise is heavy-tailed. In particular, the heavy-tail index (or heavy-tailedness itself) of the gradient noise will not be affected by hyperparameters such as the learning rate.

To be specific, Hodgkinson & Mahoney (2020); Gurbuzbalaban et al. (2021) show that heavy tails can arise in the stationary distribution of SGD through multiplicative dynamics, and the tail index of the resulting stationary distribution can be characterized by the learning rates and the magnitude of noises (through batch size). Similarly, the SDEs studied in Wojtowytsch (2021); Mori et al. (2021) are both driven by light-tailed (Gaussian) perturbations, and the authors show that even under light-tailed perturbations, heavy tails can arise in the stationary distribution through multiplicative dynamics, and the tail index of the resulting stationary distribution can be characterized by the learning rates. In comparison, we focus on the impact of the heavy-tails in the gradient noise and the truncation scheme on the global dynamics of SGD. Our results show that, under heavy-tailed noises, the power-law

index $\alpha$ for the tail in noises also characterizes the first exit time and global dynamics of SGD with no dependence on other training hyperparameters (at least in the asymptotic scheme). In other words, when the driving force of the dynamics is heavy-tailed, its "heavy-tailedness" (i.e., the same index $\alpha$) characterizes the "heavy-tailedness" of the entire SGD without dependency on the other hyperparameters, and its effect on the entire SGD path (rather than just the stationary distributions) admits a very clear and tight expression, as characterized in theoretical results in this work.

## G   PROOF OF THEOREM 1

This section proves Theorem 1. We first start with providing the definitions of $q_i$ and $q_{i,j}$ that appear in the statement of the theorem. Let $\mathbf{Leb}_+$ denote the Lebesgue measure restricted on $[0, \infty)$, and define a (Borel) measure $\nu_\alpha$ on $\mathbb{R}\backslash\{0\}$ as follows:

$$\nu_\alpha(dx) = \mathbb{1}\{x > 0\}\frac{\alpha p_+}{x^{\alpha+1}} + \mathbb{1}\{x < 0\}\frac{\alpha p_-}{|x|^{\alpha+1}}$$

where $\alpha$, $p_-$, and $p_+$ are constants in Assumption 2. Define a Borel measure $\mu_i$ on $\mathbb{R}^{l_i^*} \times \left(\mathbb{R}_+\right)^{l_i^*-1}$ as the product measure

$$\mu_i = (\nu_\alpha)^{l_i^*} \times (\mathbf{Leb}_+)^{l_i^*-1}. \tag{G.1}$$

We also define mappings $h_i$ as follows. For a real sequence $\mathbf{w} = (w_1, w_2, \cdots, w_{l_i^*})$ and a positive real number sequence $\mathbf{t} = (t'_j)_{j=2}^{l_i^*}$, define $t_1 = t'_1 = 0$ and $t_j = t'_1 + t'_2 + \cdots + t'_j$ for $j = 2, \cdots, l_i^*$. Now we define a path $\hat{\mathbf{x}} : [0, \infty) \mapsto \mathbb{R}$ as the solution to the following ODE with jumps:

$$\hat{\mathbf{x}}(0) = \varphi_L\big(m_i + \varphi_b(w_1)\big); \tag{G.2}$$

$$\frac{d\hat{\mathbf{x}}(t)}{dt} = -f'(\hat{\mathbf{x}}(t)), \quad \forall t \in [t_{j-1}, t_j), \quad \forall j = 2, \cdots, l_i^*; \tag{G.3}$$

$$\hat{\mathbf{x}}(t_j) = \varphi_L\big(\hat{\mathbf{x}}(t_j-) + \varphi_b(w_j)\big), \quad \forall j = 2, \cdots, l_i^*. \tag{G.4}$$

Now we define the mappings $h_i : \mathbb{R}^{l_i^*} \times \left(\mathbb{R}_+\right)^{l_i^*-1} \mapsto \mathbb{R}$ as

$$h_i(\mathbf{w}, \mathbf{t}) = \hat{\mathbf{x}}(t_{l_i^*}).$$

It is easy to see that $h_i$'s are continuous mappings. With these mappings, we define the following sets:

$$E_i \triangleq \{(\mathbf{w}, \mathbf{t}) \subseteq \mathbb{R}^{l_i^*} \times \left(\mathbb{R}_+\right)^{l_i^*-1} : h_i(\mathbf{w}, \mathbf{t}) \notin \Omega_i\}; \tag{G.5}$$

$$E_{i,j} \triangleq \{(\mathbf{w}, \mathbf{t}) \subseteq \mathbb{R}^{l_i^*} \times \left(\mathbb{R}_+\right)^{l_i^*-1} : h_i(\mathbf{w}, \mathbf{t}) \in \Omega_j\}. \tag{G.6}$$

Lastly, the constant $q_i$ and $q_{i,j}$ are defined as follows:

$$q_i = \mu_i(E_i), \quad q_{i,j} = \mu_i(E_{i,j}) \ \forall i \neq j. \tag{G.7}$$

Before we move on to the proof of Theorem 1, we add a few remarks regarding the intuition behind it.

- Suppose that $X_n^\eta$ is started at the $i^{\text{th}}$ local minimum $m_i$ of $f$, and consider the behavior of $X_n^\eta$ over the time period $H_1 \triangleq \{1, \ldots, \lceil t/\eta \rceil\}$ for a sufficiently large $t$. The heavy-tailed large deviations theory Rhee et al. (2019) and a heuristic application of the contraction principle implies that the path of $X_{\lceil n/\eta \rceil}^\eta$ over this period will converge to the gradient flow of $f$, and the event that $X_{\lceil n/\eta \rceil}^\eta$ escapes $\Omega_i$ within this period is a (heavy-tailed) rare event. This means that the probability of such an event is of order $(1/\eta)^{(\alpha-1)l_i^*}$. Moreover, whenever it happens, it is almost always because $X_n^\eta$ is shaken by *exactly $l_i^*$ large gradient noises of size $\mathcal{O}(1/\eta)$*, which translates to $l_i^*$ jumps in $X_{\lceil n/\eta \rceil}^\eta$'s path, while the rest of its path closely resemble the deterministic gradient flow. Moreover, conditional on the event

that $X_n^\eta$ fails to escape from $\Omega_i$ within this period, the endpoint of the path is most likely to be close to the local minima, i.e., $X_{\lceil t/\eta \rceil}^\eta \approx m_i$. This suggests that over the next time period $H_2 \triangleq \{\lceil t/\eta \rceil + 1, \lceil t/\eta \rceil + 2, \ldots, 2\lceil t/\eta \rceil\}$ of length $\lceil t/\eta \rceil$, $X_n^\eta$ will behave similarly to its behavior over the first period $H_1$. The same argument applies to the subsequent periods $H_3, H_4, \ldots$ as well. Therefore, over each time period of length $\lceil t/\eta \rceil$, there is $(1/\eta)^{(\alpha-1)l_i^*}$ probability of exit. In view of this, the exit time should be of order $(1/\eta)^{1+(\alpha-1)l_i^*}$ and resemble an exponential distribution when scaled properly.

- Part (i) of Theorem 1 builds on this intuition and rigorously prove that the first exit time is indeed roughly of order $1/\lambda_i(\eta) \approx (1/\eta)^{1+(\alpha-1)l_i^*}$ and resembles an exponential distribution.

- Given this, one would expect that $X_{\sigma_i(\eta)}^\eta$, the location of SGD right at the time of exit, will hardly ever be farther than $l_i^* b$ away from $m_i$: the length of each update is clipped by $b$, and there will most likely be only $l_i^*$ large SGD steps during this successful attempt. Indeed, from the definition of $q_{i,j}$'s above, one can see that $q_{i,j} > 0$ if and only if $\inf_{y \in \Omega_j} |y - m_i| < l_i^* b$.

Summarizing the three bullet points here, we see that the minimum number of jumps $l_i^*$ dictates *how* heavy-tailed SGD escapes an attraction field, *where* the SGD lands on upon its exit, and *when* the exit occurs.

Now we are ready to start proving Theorem 1. First, note that Assumption 1 implies the following:

- There exist $c_0 > 0, \epsilon_0 \in (0, 1)$ such that for any $x \in \{m_1, s_1, \cdots, s_{n_{\min}-1}, m_{n_{\min}}\}, |y-x| < \epsilon_0$,

$$|f'(y)| > c_0 |y - x|, \tag{G.8}$$

and for any $y \in [-L, L]$ such that $|y - x| \geq \epsilon_0$ for all $x \in \{m_1, s_1, \cdots, s_{n_{\min}-1}, m_{n_{\min}}\}$, we have

$$|f'(y)| > c_0; \tag{G.9}$$

- There exist constants $L \in (0, \infty), M \in (0, \infty)$ such that $|m_0| < L, |m_{n_{\min}}| < L$, and (for any $x \in [-L, L]$)

$$|f'(x)| \leq M, \ |f''(x)| \leq M. \tag{G.10}$$

Recall that for $c > 0$, the truncation operator was defined as

$$\varphi_c(w) \triangleq \varphi(w, c) \triangleq (w \wedge c) \vee (-c), \quad \forall w \in \mathbb{R}, \tag{G.11}$$

and the SGD iterates were defined as

$$X_n^\eta = \varphi_L\Big(X_n^\eta - \varphi_b\big(\eta(f'(X_n^\eta) - Z_{n+1})\big)\Big). \tag{G.12}$$

Here $\eta > 0$ is the learning rate (step length) and $b > 0$ is the gradient clipping threshold. Also, recall that for any $k \in [n_{\min}]$, we let

$$\sigma_k(\eta) \triangleq \min\{n \geq 0 : X_n^\eta \notin \Omega_k\}$$

to be the time that $X_n^\eta$ exists from the $k-$th attraction field $\Omega_j$. Meanwhile, given the gradient clipping threshold $b$, recall that

$$r_i \triangleq \min\{m_i - s_{i-1}, s_i - m_i\}, \tag{G.13}$$
$$l_i^* \triangleq \lceil r_i/b \rceil. \tag{G.14}$$

Intuitively speaking, $l_i^*$ tells us the minimum number of jumps with size no larger than $b$ required in order to escape the attraction field if we start from the local minimum of this attraction field $\Omega_i$. Lastly, recall the definition of $H(\cdot) = \mathbb{P}(|Z_1| > \cdot)$ and

$$\lambda_i(\eta) \triangleq H(1/\eta)\Big(\frac{H(1/\eta)}{\eta}\Big)^{l_i^*-1}.$$

The proof of Theorem 1 hinges on the following two lemmas that characterize the behavior of $X_n^\eta$ in two different phases respectively. Let $k \in [n_{\min}]$ and $x \in \Omega_k$. We consider the SGD iterates initialized at $X_0^\eta = x$. In the first phase, the SGD iterates return to $[m_k - 2\epsilon, m_k + 2\epsilon]$, a small neighborhood of the local minimizer in attraction field $\Omega_k$; in other words, it ends at

$$T_{\text{return}}^{(k)}(\eta, \epsilon) \triangleq \min\{n \geq 0 : X_n^\eta \in [m_k - 2\epsilon, m_k + 2\epsilon]\}. \tag{G.15}$$

During this phase, we show that for all learning rate $\eta$ that is sufficiently small, it is almost always the case that $X_n^\eta$ would quickly return to $[m_k - 2\epsilon, m_k + 2\epsilon]$, and it never leaves $\Omega_k$ before $T_{\text{return}}^{(k)}$.

**Lemma G.1.** *There exists some $c \in (0, \infty)$ such that for any $k \in [n_{min}]$, the following claim holds for all $\epsilon > 0$ small enough:*

$$\lim_{\eta \downarrow 0} \inf_{y \in [-L,L]:\ y \in (s_{k-1}+\epsilon, s_k-\epsilon)} \mathbb{P}_y\Big( X_n^\eta \in \Omega_k\ \forall n \in \big[T_{return}^{(k)}(\eta, \epsilon)\big],\ T_{return}^{(k)}(\eta, \epsilon) \leq \frac{c\log(1/\epsilon)}{\eta} \Big) = 1.$$

During the second phase, $X_n^\eta$ starts from somewhere in $[m_k - 2\epsilon, m_k + 2\epsilon]$ and tries to escape from $\Omega_k$, meaning that the phase ends at $\sigma_k(\eta)$. During this phase, we show that the distributions of the first exit time $\sigma_k(\eta)$ and the location $X_{\sigma_k(\eta)}^\eta$ do converge to the ones described in Theorem 1 as learning rate $\eta$ tends to 0.

**Lemma G.2.** *There exist constants $q_i > 0\ \forall i \in [n_{min}]$ and $q_{i,j} \geq 0\ \forall j \in [n_{min}]$ with $i, j \neq i$ such that the following claim holds: given any $C > 0, u > 0$ and any $k, l \in [n_{min}]$ with $k \neq l$, we have*

$$\limsup_{\eta \downarrow 0} \sup_{x \in [-L,L],\ x \in (m_k-2\epsilon, m_k+2\epsilon)} \mathbb{P}_x\Big( q_k\lambda_k(\eta)\sigma_k(\eta) > u \Big) \leq C + \exp\big(-(1-C)u\big) \tag{G.16}$$

$$\liminf_{\eta \downarrow 0} \inf_{x \in [-L,L],\ x \in (m_k-2\epsilon, m_k+2\epsilon)} \mathbb{P}_x\Big( q_k\lambda_k(\eta)\sigma_k(\eta) > u \Big) \geq -C + \exp\big(-(1+C)u\big) \tag{G.17}$$

$$\limsup_{\eta \downarrow 0} \sup_{x \in [-L,L],\ x \in (m_k-2\epsilon, m_k+2\epsilon)} \mathbb{P}_x\Big( X_{\sigma_k(\eta)}^\eta \in \Omega_l \Big) \leq \frac{q_{k,l} + C}{q_k} \tag{G.18}$$

$$\liminf_{\eta \downarrow 0} \inf_{x \in [-L,L],\ x \in (m_k-2\epsilon, m_k+2\epsilon)} \mathbb{P}_x\Big( X_{\sigma_k(\eta)}^\eta \in \Omega_l \Big) \geq \frac{q_{k,l} - C}{q_k} \tag{G.19}$$

*for all $\epsilon > 0$ that are sufficiently small.*

Now we are ready to show Theorem 1.

*Proof of Theorem 1.* Fix some $k \in [n_{\min}]$ and $x \in \Omega_k \cap [-L, L]$. Let $q_k$ and $q_{k,l}$ be the constants in Lemma G.2.

We first prove the weak convergence claim in Theorem 1(i). Arbitrarily choose some $u > 0$ and $C \in (0, 1)$. It suffices to show that

$$\limsup_{\eta \downarrow 0} \mathbb{P}_x(q_k\lambda_k(\eta)\sigma_k(\eta) > u) \leq 2C + \exp\big(-(1-C)u\big),$$

$$\liminf_{\eta \downarrow 0} \mathbb{P}_x(q_k\lambda_k(\eta)\sigma_k(\eta) > u) \geq (1-C)\Big(-C + \exp\big(-(1+C)u\big)\Big).$$

Recall the definition of the stopping time $T_{\text{return}}^{(k)}$ in eq. (G.15). Define event

$$A_k(\eta, \epsilon) \triangleq \Big\{ X_n^\eta \in \Omega_k\ \forall n \in \big[T_{\text{return}}^{(k)}(\eta, \epsilon)\big],\ T_{\text{return}}^{(k)}(\eta, \epsilon) \leq \frac{c\log(1/\epsilon)}{\eta} \Big\}$$

where $c < \infty$ is the constant in Lemma G.1. First, since $x \in \Omega_k = (s_{k-1}, s_k)$, it holds for all $\epsilon > 0$ small enough that $x \in (s_{k-1} + \epsilon, s_k - \epsilon)$. Next, one can find some $\epsilon > 0$ such that

- (Due to Lemma G.1)

$$\mathbb{P}_x\big((A_k(\eta, \epsilon))^c\big) \leq C\ \ \forall \eta \text{ sufficiently small};$$

- (Due to eq. (G.16)eq. (G.17) and strong Markov property) For all $\eta$ sufficiently small,

$$\mathbb{P}_x\Big( q_k\lambda_k(\eta)\big(\sigma(\eta) - T_{\text{return}}^{(k)}(\eta, \epsilon)\big) > (1-C)u\ \Big|\ A_k(\eta, \epsilon) \Big) \leq C + \exp\big(-(1-C)u\big),$$

$$\mathbb{P}_x\Big( q_k\lambda_k(\eta)\big(\sigma(\eta) - T_{\text{return}}^{(k)}(\eta, \epsilon)\big) > u\ \Big|\ A_k(\eta, \epsilon) \Big) \geq -C + \exp\big(-(1+C)u\big).$$

Fix such $\epsilon$. Lastly, for this fixed $\epsilon$, due to $\lambda_k \in \mathcal{RV}_{-1-l_k^*(\alpha-1)}(\eta)$ and $\alpha > 1$, we have $q_k\lambda_k(\eta) \cdot \frac{c\log(1/\epsilon)}{\eta} < Cu$ for all $\eta$ sufficiently small. In summary, for all $\eta$ sufficiently small, we have

$$\mathbb{P}_x(q_k\lambda_k(\eta)\sigma_k(\eta) > u)$$

$$\leq \mathbb{P}_x\big((A_k(\eta,\epsilon))^c\big) + \mathbb{P}_x\Big(\big\{q_k\lambda_k(\eta)\sigma_k(\eta) > u\big\} \cap A_k(\eta,\epsilon)\Big)$$

$$\leq C + \mathbb{P}_x\Big(\big\{q_k\lambda_k(\eta)\sigma_k(\eta) > u\big\} \cap A_k(\eta,\epsilon)\Big)$$

$$= C + \mathbb{P}_x\Big(q_k\lambda_k(\eta)\big(\sigma(\eta) - T_{\text{return}}^{(k)}(\eta,\epsilon)\big) > u - q_k\lambda_k(\eta)T_{\text{return}}^{(k)}(\eta,\epsilon) \ \Big|\ A_k(\eta,\epsilon)\Big) \cdot \mathbb{P}_x(A_k(\eta,\epsilon))$$

$$\leq C + \mathbb{P}_x\Big(q_k\lambda_k(\eta)\big(\sigma(\eta) - T_{\text{return}}^{(k)}(\eta,\epsilon)\big) > (1-C)u \ \Big|\ A_k(\eta,\epsilon)\Big)$$

$$\leq 2C + \exp\big(-(1-C)u\big)$$

and

$$\mathbb{P}_x(q_k\lambda_k(\eta)\sigma_k(\eta) > u)$$

$$\geq \mathbb{P}_x\Big(\big\{q_k\lambda_k(\eta)\sigma_k(\eta) > u\big\} \cap A_k(\eta,\epsilon)\Big)$$

$$= \mathbb{P}_x\Big(q_k\lambda_k(\eta)\big(\sigma(\eta) - T_{\text{return}}^{(k)}(\eta,\epsilon)\big) > u - q_k\lambda_k(\eta)T_{\text{return}}^{(k)}(\eta,\epsilon) \ \Big|\ A_k(\eta,\epsilon)\Big) \cdot \mathbb{P}_x(A_k(\eta,\epsilon))$$

$$\geq \mathbb{P}_x\Big(q_k\lambda_k(\eta)\big(\sigma(\eta) - T_{\text{return}}^{(k)}(\eta,\epsilon)\big) > u - q_k\lambda_k(\eta)T_{\text{return}}^{(k)}(\eta,\epsilon) \ \Big|\ A_k(\eta,\epsilon)\Big) \cdot (1-C)$$

$$\geq \mathbb{P}_x\Big(q_k\lambda_k(\eta)\big(\sigma(\eta) - T_{\text{return}}^{(k)}(\eta,\epsilon)\big) > u \ \Big|\ A_k(\eta,\epsilon)\Big) \cdot (1-C)$$

$$\geq (1-C)\Big(-C + \exp\big(-(1+C)u\big)\Big)$$

so this concludes the proof for Theorem 1(i).

In order to prove claims in Theorem 1(ii), we first observe that on event $A_k(\eta,\epsilon)$ we must have $\sigma(\eta) > T_{\text{return}}^{(k)}(\eta,\epsilon)$. Next, arbitrarily choose some $C \in (0,1)$, and note that it suffices to show that $\mathbb{P}_x(X_{\sigma(\eta)}^\eta \in \Omega_l) \in \Big((1-C)\frac{q_{k,l}-C}{q_k}, C + \frac{q_{k,l}+C}{q_k}\Big)$ holds for all $\eta$ sufficiently small. Again, we can find $\epsilon > 0$ such that

- (Due to Lemma G.1)
$$\mathbb{P}_x\big((A_k(\eta,\epsilon))^c\big) \leq C \ \ \forall \eta \text{ sufficiently small;}$$

- (Due to eq. (G.18)eq. (G.19) and strong Markov property) For all $\eta$ sufficiently small,
$$\frac{q_{k,l}-C}{q_k} \leq \mathbb{P}_x\Big(X_{\sigma_k(\eta)}^\eta \in \Omega_l \ \Big|\ A_k(\eta,\epsilon)\Big) \leq \frac{q_{k,l}+C}{q_k}.$$

In summary, for all $\eta$ sufficiently small, we have

$$\mathbb{P}_x\Big(X_{\sigma_k(\eta)}^\eta \in \Omega_l\Big) \leq \mathbb{P}_x\big((A_k(\eta,\epsilon))^c\big) + \mathbb{P}_x\Big(\big\{X_{\sigma_k(\eta)}^\eta \in \Omega_l\big\} \cap A_k(\eta,\epsilon)\Big)$$

$$\leq C + \mathbb{P}_x\Big(X_{\sigma_k(\eta)}^\eta \in \Omega_l \ \Big|\ A_k(\eta,\epsilon)\Big)\mathbb{P}_x(A_k(\eta,\epsilon))$$

$$\leq C + \frac{q_{k,l}+C}{q_k}$$

and

$$\mathbb{P}_x\Big(X_{\sigma_k(\eta)}^\eta \in \Omega_l\Big) \geq \mathbb{P}_x\Big(\big\{X_{\sigma_k(\eta)}^\eta \in \Omega_l\big\} \cap A_k(\eta,\epsilon)\Big)$$

$$= \mathbb{P}_x\Big(X_{\sigma_k(\eta)}^\eta \in \Omega_l \ \Big|\ A_k(\eta,\epsilon)\Big)\mathbb{P}_x(A_k(\eta,\epsilon))$$

$$\geq (1-C)\frac{q_{k,l}-C}{q_k}$$

and this concludes the proof. $\qquad\square$

The rest of this section is devoted to the proofs of Lemma G.1 and Lemma G.2. Specifically, Lemma G.1 is an immediate Corollary of Lemma G.13, the proof of which will be provided below. The proof of Lemma G.2 can be found at the end of this section.

## G.1 Proofs of Lemma G.1, G.2

The following three lemmas will be applied repeatedly throughout this section. The proofs are straightforward but provided in Section K for the sake of completeness.

**Lemma G.3.** *Given two real functions* $a : \mathbb{R}_+ \mapsto \mathbb{R}_+$, $b : \mathbb{R}_+ \mapsto \mathbb{R}_+$ *such that* $a(\epsilon) \downarrow 0, b(\epsilon) \downarrow 0$ *as* $\epsilon \downarrow 0$, *and a family of geometric RVs* $\{U(\epsilon) : \epsilon > 0\}$ *with success rate* $a(\epsilon)$ *(namely,* $\mathbb{P}(U(\epsilon) > k) = (1 - a(\epsilon))^k$ *for* $k \in \mathbb{N}$*), for any* $c > 1$*, there exists* $\epsilon_0 > 0$ *such that for any* $\epsilon \in (0, \epsilon_0)$,

$$\exp\left(-\frac{c \cdot a(\epsilon)}{b(\epsilon)}\right) \leq \mathbb{P}\left(U(\epsilon) > \frac{1}{b(\epsilon)}\right) \leq \exp\left(-\frac{a(\epsilon)}{c \cdot b(\epsilon)}\right).$$

**Lemma G.4.** *Given two real functions* $a : \mathbb{R}_+ \mapsto \mathbb{R}_+$, $b : \mathbb{R}_+ \mapsto \mathbb{R}_+$ *such that* $a(\epsilon) \downarrow 0, b(\epsilon) \downarrow 0$ *and*

$$a(\epsilon)/b(\epsilon) \to 0$$

*as* $\epsilon \downarrow 0$, *and a family of geometric RVs* $\{U(\epsilon) : \epsilon > 0\}$ *with success rate* $a(\epsilon)$ *(namely,* $\mathbb{P}(U(\epsilon) > k) = (1 - a(\epsilon))^k$ *for* $k \in \mathbb{N}$*), for any* $c > 1$ *there exists some* $\epsilon_0 > 0$ *such that for any* $\epsilon \in (0, \epsilon_0)$,

$$a(\epsilon)/(c \cdot b(\epsilon)) \leq \mathbb{P}(U(\epsilon) \leq 1/b(\epsilon)) \leq c \cdot a(\epsilon)/b(\epsilon)$$

**Lemma G.5.** *Suppose that a function* $g : E \mapsto \mathbb{R}$ *(where* $E$ *is an open set in* $\mathbb{R}^d$*) is* $g \in \mathcal{C}^2(E)$ *and* $\|\nabla^2 g(\cdot)\| \leq C$ *on its domain* $E$ *for some constant* $C < \infty$*. For a finite integer* $n$*, a sequence of vectors* $\{z_1, \cdots, z_n\}$ *in* $\mathbb{R}^d$*, and vectors* $x, \widetilde{x} \in E, \eta > 0$*, consider two sequences* $\{x_k\}_{k=0,\ldots,n}$ *and* $\{\widetilde{x}_k\}_{k=0,\ldots,n}$ *constructed as follows:*

$$\begin{aligned}
x_0 &= x \\
x_k &= x_{k-1} + \eta \nabla g(x_{k-1}) + \eta z_k \quad \forall k = 1, 2, \cdots, n \\
\widetilde{x}_0 &= \widetilde{x} \\
\widetilde{x}_k &= \widetilde{x}_{k-1} + \eta \nabla g(\widetilde{x}_{k-1}) \quad \forall k = 1, 2, \cdots, n
\end{aligned}$$

*If we have that the line segment from* $x_k$ *to* $\widetilde{x}_k$ *is contained in* $E$ *and* $\eta \|z_1 + \cdots + z_k\| + \|x - \widetilde{x}\| \leq \widetilde{c}$ *for all* $k = 1, 2, \cdots, n$ *for some* $\widetilde{c} > 0$*, then*

$$\|x_k - \widetilde{x}_k\| \leq \widetilde{c} \cdot \exp(\eta C k) \quad \forall k = 1, 2, \cdots, n.$$

To facilitate the analysis below, we introduce some additional notations. First, we will group the noises $Z_n$ based on a threshold level $\delta > 0$: let us define

$$Z_n^{\leq \delta, \eta} \triangleq Z_n \mathbb{1}\{\eta |Z_n| \leq \delta\}, \tag{G.20}$$

$$Z_n^{>\delta, \eta} \triangleq Z_n \mathbb{1}\{\eta |Z_n| > \delta\}. \tag{G.21}$$

The former are viewed as *small noises* while the latter will be referred to as *large noises* or *large jumps*. Furthermore, for any $j \geq 1$, define the $j^{\text{th}}$ arrival time and size of large jumps as

$$T_j^\eta(\delta) \triangleq \min\{n > T_{j-1}^\eta(\delta) : \eta |Z_n| > \delta\}, \quad T_0^\eta(\delta) = 0 \tag{G.22}$$

$$W_j^\eta(\delta) \triangleq Z_{T_j^\eta(\delta)}. \tag{G.23}$$

Next, for any $\epsilon > 0$, let $\Omega_i(\epsilon) = [m_i - \epsilon, m_i + \epsilon]$ be an $\epsilon$−neighborhood of the local minimum $m_i$, and $S_i(\epsilon) = [s_i - \epsilon, s_i + \epsilon]$ be an $\epsilon$−neighborhood of the local maximum $s_i$.

For most part of this section, we will zoom in on one of the local minima $m_i$ and its attraction field $\Omega_i = (s_{i-1}, s_i)$. Without loss of generality, we assume $m_i = 0$, and denote the attraction field as $\Omega = (s_-, s_+)$. (If $m_i$ happens to be the local minimum at the left or right boundary, then the attraction field is $[-L, s_+)$ or $(s_-, L]$ where the SGD iterates will be reflected at $\pm L$.) Henceforth we will drop the dependency on notation $i$ when referring to this specific attraction field until the very end of this section. Throughout the proof, the following (deterministic) dynamic systems will be used frequently as benchmark processes to indicate the most likely location of the SGD iterates. Specifically, given

any $x \in \Omega$, we use $X_n(x)$ to indicate that the starting point is $x$, namely $X_0(x) = x$. Similarly, consider the following ODE $\mathbf{x}^\eta(t; x)$ as

$$\mathbf{x}^\eta(0; x) = x; \tag{G.24}$$

$$\frac{d\mathbf{x}^\eta(t; x)}{dt} = -\eta f'\Big(\mathbf{x}^\eta(t; x)\Big). \tag{G.25}$$

When we use update rate $\eta = 1$, we will drop the dependency of $\eta$ and simply use $\mathbf{x}(t; x)$ to denote the process.

Based on Assumption 3, we know the existence of some constant $\bar{\epsilon} \in (0, \epsilon_0)$ (note that $\epsilon_0$ is the constant in eq. (G.8)) such that

$$r \triangleq \min\{-s_-, s_+\}, \tag{G.26}$$

$$l^* \triangleq \lceil r/b \rceil, \tag{G.27}$$

$$(l^* - 1)b + 100l^*\bar{\epsilon} < r - 100l^*\bar{\epsilon} \tag{G.28}$$

$$r + 100l^*\bar{\epsilon} < l^*b - 100l^*\bar{\epsilon}. \tag{G.29}$$

Here $r$ can be understood as the effective *radius* of the said attraction field. Also, we fix such $\bar{\epsilon}$ small enough so that (let $c_-^L = -f'(-L), c_+^L = -f'(-L)$), we have

$$0.9c_-^L \leq -f'(x) \leq 1.1c_-^L \ \ \forall x \in [-L, -L + 100\bar{\epsilon}], \tag{G.30}$$

$$0.9c_+^L \geq -f'(x) \geq 1.1c_+^L \ \ \forall x \in [L - 100\bar{\epsilon}, L]. \tag{G.31}$$

Similar to the definition of ODE $\mathbf{x}^\eta$, let us consider the following construction of ODE $\tilde{\mathbf{x}}^\eta$ that can be understood as $\mathbf{x}^\eta$ perturbed by $l^*$ shocks. Specifically, consider a sequence or real numbers $0 = t_1 < t_2 < t_3 < \cdots < t_{l^*}$ and real numbers $w_1, \cdots, w_{l^*}$ where $|w_j| \leq b$ for each $j$. Let $\mathbf{t} = (t_1, \cdots, t_{l^*}), \mathbf{w} = (w_1, \cdots, w_{l^*})$. Based on these two sequences and rate $\eta > 0$, define $\tilde{\mathbf{x}}^\eta(t; x)$ as

$$\tilde{\mathbf{x}}^\eta(0, x; \mathbf{t}, \mathbf{w}) = \varphi_L(x + \varphi_b(w_1)); \tag{G.32}$$

$$\frac{d\tilde{\mathbf{x}}^\eta(t, x; \mathbf{t}, \mathbf{w})}{dt} = -\eta f'\big(\tilde{\mathbf{x}}^\eta(t, x; \mathbf{t}, \mathbf{w})\big) \ \ \forall t \notin \{t_1, t_2, \cdots, t_{l^*}\} \tag{G.33}$$

$$\tilde{\mathbf{x}}^\eta(t_j, x; \mathbf{t}, \mathbf{w}) = \varphi_L\big(\tilde{\mathbf{x}}^\eta(t_j-, x; \mathbf{t}, \mathbf{w}) + \varphi_b(w_j)\big) \ \ \forall j = 2, \cdots, l^* \tag{G.34}$$

Again, when $\eta = 1$ we drop the notational dependency on $\eta$ and use $\tilde{\mathbf{x}}$ to denote the process. Now from eq. (G.28)eq. (G.29) one can easily see the following fact: there exist constants $\bar{t}, \bar{\delta} > 0$ such that $\tilde{\mathbf{x}}(t_{l^*}, 0; \mathbf{t}, \mathbf{w}) \notin \Omega$ (note that the starting point is 0, the local minimum) only if (under the condition that $|w_j| \leq b \ \forall j$)

$$t_j - t_{j-1} \leq \bar{t} \ \ \forall j = 2, 3, \cdots, l^* \tag{G.35}$$

$$|w_j| > \bar{\delta} \ \ \forall j = 1, 2, \cdots, l^*. \tag{G.36}$$

The intuition is as follows: if the inter-arrival time between any of the $l^*$ jumps is too long, then the path of $\tilde{\mathbf{x}}^\eta(t; x)$ will drift back to the local minimum $m_i$ so that the remaining $l^* - 1$ shocks (whose sizes are bounded by $b$) cannot overcome the radius $r$ which is strictly larger than $(l^* - 1)b$; similarly, if size of any of the shocks is too small, then since all other jumps have sizes bounded by $b$, the shock created by the $l_i^*$ jumps will be smaller than $(l^* - 1)b + 100\bar{\epsilon}$, which is strictly less than $r$. We fix these constants $\bar{t}, \bar{\delta}$ throughout the analysis, and stress again that their values are dictated by the geometry of the function $f$, thus *do not* vary with the accuracy parameters $\epsilon$ and $\delta$ mentioned earlier. In particular, choose $\bar{\delta}$ such that $\bar{\delta} < \bar{\epsilon}$.

In our analysis below, $\epsilon > 0$ will be a variable representing the level of *accuracy* in our analysis. For instance, for small $\epsilon$, the chance that SGD iterates will visit somewhere that is $\epsilon$-close to $s_-$ or $s_+$ (namely, the boundary of the attraction filed) should be small. Consider some $\epsilon \in (0, \epsilon_0)$ where $\epsilon_0$ is the constant in Assumption 1. Due to eq. (G.8)eq. (G.9), one can see the existence of some $g_0 > 0, c_1 < \infty$ such that

- $|f'(x)| \geq g_0$ for any $x \in \Omega$ such that $|x - s_-| > \epsilon_0, ||x - s_+| > \epsilon_0$;

- Let $\hat{t}_{\text{ODE}}(x, \eta) \triangleq \min\{t \geq 0 : \mathbf{x}^\eta(t, x) \in [-\epsilon, \epsilon]\}$ be the time that the ODE returns to a $\epsilon$−neighborhood of local minimum of $\Omega$ when starting from $x$. As proved in Lemma 3.5 of Pavlyukevich (2005), for any $x \in \Omega$ such that $|x - s_-| > \epsilon, |x - s_+| > \epsilon$, we have

$$\hat{t}_{\text{ODE}}(x, \eta) \leq c_1 \frac{\log(1/\epsilon)}{\eta} \qquad (\text{G.37})$$

and we define the function

$$\hat{t}(\epsilon) \triangleq c_1 \log(1/\epsilon). \qquad (\text{G.38})$$

In short, given any accuracy level $\epsilon$, the results above give us an upper bound for how fast the ODE would return to a neighborhood of the local minimum, if the starting point is not too close to the boundary of this attraction field $\Omega$.

For the first few technical results established below, we show that, without large jumps, the SGD iterates $X_n^\eta(x)$ are unlikely to show significant deviation from the deterministic gradient descent process $\mathbf{y}_n^\eta(x)$ defined as

$$\mathbf{y}_0^\eta(x) = x, \qquad (\text{G.39})$$

$$\mathbf{y}_n^\eta(x) = \mathbf{y}_{n-1}^\eta(x) - \eta f'\left(\mathbf{y}_{n-1}^\eta(x)\right). \qquad (\text{G.40})$$

We are ready to state the first lemma, where we bound the distance between the gradient descent iterates $\mathbf{y}_n^\eta(y)$ and the ODE $\mathbf{x}^\eta(t, x)$ when the initial conditions $x, y$ are close enough.

**Lemma G.6.** *The following claim holds for all $\eta > 0$: for any $t > 0$, we have*

$$\sup_{s \in [0,t]} |\mathbf{x}^\eta(s, x) - \mathbf{y}_{\lfloor s \rfloor}^\eta(y)| \leq (2\eta M + |x - y|) \exp(\eta M t)$$

*where $M \in (0, \infty)$ is the constant in eq. (G.10).*

*Proof.* Define a continuous-time process $\mathbf{y}^\eta(s; y) \triangleq \mathbf{y}_{\lfloor s \rfloor}^\eta(y)$, and note that

$$\mathbf{x}^\eta(s, x) = \mathbf{x}^\eta(\lfloor s \rfloor, x) - \eta \int_{\lfloor s \rfloor}^s f'(\mathbf{x}^\eta(u, x)) du$$

$$\mathbf{x}^\eta(\lfloor s \rfloor, x) = x - \eta \int_0^{\lfloor s \rfloor} f'(\mathbf{x}^\eta(u, x)) du$$

$$\mathbf{y}_{\lfloor s \rfloor}^\eta(y) = \mathbf{y}^\eta(\lfloor s \rfloor, y) = y - \eta \int_0^{\lfloor s \rfloor} f'(\mathbf{y}^\eta(u, y)) du.$$

Therefore, if we define function

$$b(u) = \mathbf{x}^\eta(u, x) - \mathbf{y}^\eta(u, y),$$

from the fact $|f'(\cdot)| \leq M$, one can see that $|b(u)| \leq \eta M + |x - y|$ for any $u \in [0, 1)$ and $|b(1)| \leq 2\eta M + |x - y|$. In case that $s > 1$, from the display above and the fact $|f''(\cdot)| \leq M$, we now have

$$|\mathbf{y}_{\lfloor s \rfloor}^\eta(x) - \mathbf{x}^\eta(s, x)| \leq |b(\lfloor s \rfloor)| + \eta M;$$

$$|b(\lfloor s \rfloor)| \leq \eta M \int_1^{\lfloor s \rfloor} |b(u)| du.$$

From Gronwall's inequality (see Theorem 68, Chapter V of Protter (2005), where we let function $\alpha(u)$ be $\alpha(u) = |b(u+1)|$), we have

$$|\mathbf{y}_{\lfloor s \rfloor}^\eta(x) - \mathbf{x}^\eta(s, x)| \leq (2\eta M + |x - y|) \exp(\eta M t).$$

This concludes the proof. $\qquad \square$

Now we consider an extension of the previous Lemma in the following sense: we add perturbations to the gradient descent process and ODE, and show that, when both perturbed by $l^*$ similar perturbations, the ODE and gradient descent process should still stay close enough. Analogous to the definition of the perturbed ODE $\widetilde{x}^\eta$ in eq. (G.32)-eq. (G.34), we can construct a process $\widetilde{Y}^\eta$ as a perturbed gradient descent process as follows. For a sequence of integers $0 = t_1 < t_2 < \cdots < t_{l^*}$ (let $\mathbf{t} = (t_j)_{j \geq 1}$) and a sequence of real numbers $\widetilde{w}_1, \cdots, \widetilde{w}_{l^*}$ (let $\widetilde{\mathbf{w}} = (\widetilde{w}_j)_{j \geq 1}$) and $y \in \mathbb{R}$, define (for all $n = 1, 2, \cdots, t_{l^*}$) the perturbed gradient descent iterates with gradient clipping at $b$ and reflection at $\pm L$ as

$$\widetilde{\mathbf{y}}_n^\eta(y; \mathbf{t}, \widetilde{\mathbf{w}}) = \varphi_L\left(\widetilde{\mathbf{y}}_{n-1}^\eta(y; \mathbf{t}, \widetilde{\mathbf{w}}) + \varphi_b\left(-\eta f'(\widetilde{\mathbf{y}}_{n-1}^\eta(y; \mathbf{t}, \widetilde{\mathbf{w}})) + \sum_{j=2}^{l^*} \mathbb{1}\{n = t_j\}\widetilde{w}_j\right)\right) \quad \text{(G.41)}$$

with initial condition $\widetilde{\mathbf{y}}_0^\eta(y; \mathbf{t}, \widetilde{\mathbf{w}}) = \varphi_L\left(y + \varphi_b(\widetilde{w}_1)\right)$.

**Corollary G.7.** *Given any $\epsilon > 0$, the following claim holds for all sufficiently small $\eta > 0$: for any $x, y \in \Omega$, and sequence of integers $\mathbf{t} = (t_j)_{j=1}^{l^*}$ and any two sequence of real numbers $\mathbf{w} = (w_j)_{j=1}^{l^*}, \widetilde{\mathbf{w}} = (\widetilde{w}_j)_{j \geq 1}$ such that*

- $|x - y| < \epsilon$;
- $t_1 = 0$, and $t_j - t_{j-1} \leq 2\bar{t}/\eta$ for all $j \geq 1$ where $\bar{t}$ is the constant in eq. (G.35);
- $|w_j - \widetilde{w}_j| < \epsilon$ for all $j \geq 1$;

*then we have*

$$\sup_{t \in [0, t_{l^*}]} |\widetilde{\mathbf{x}}^\eta(t, x; \mathbf{t}, \mathbf{w}) - \widetilde{\mathbf{y}}_{\lfloor t \rfloor}^\eta(y; \mathbf{t}, \widetilde{\mathbf{w}})| \leq \bar{\rho}\epsilon$$

*where the constant $\bar{\rho} = (3\exp(\eta M\bar{t}) + 3)^{l^*}$.*

*Proof.* Throughout this proof, fix some $\eta \in (0, \epsilon/2M)$. We will show that for any $\eta$ in the range the claim would hold.

First, on interval $[0, t_2)$, from Lemma G.6, one can see that (since $2M\eta < \epsilon$)

$$\sup_{t \in [0, t_2)} |\widetilde{\mathbf{x}}^\eta(t, x; \mathbf{t}, \mathbf{w}) - \widetilde{\mathbf{y}}_{\lfloor t \rfloor}^\eta(y; \mathbf{t}, \widetilde{\mathbf{w}})| \leq 3\exp(\eta M\bar{t}) \cdot \epsilon.$$

The at $t = t_2$, by considering the difference between $w_2$ and $\widetilde{w}_2$, and the possible change due to one more gradient descent step (which is bounded by $\eta M < \epsilon$), we have

$$\sup_{t \in [0, t_2]} |\widetilde{\mathbf{x}}^\eta(t, x; \mathbf{t}, \mathbf{w}) - \widetilde{\mathbf{y}}_{\lfloor t \rfloor}^\eta(y; \mathbf{t}, \widetilde{\mathbf{w}})| \leq (3\exp(\eta M\bar{t}) + 2) \cdot \epsilon.$$

Now we proceed inductively. For any $j = 2, 3, \cdots, l^* - 1$, assume that

$$\sup_{t \in [0, t_j]} |\widetilde{\mathbf{x}}^\eta(t, x; \mathbf{t}, \mathbf{w}) - \widetilde{\mathbf{y}}_{\lfloor t \rfloor}^\eta(y; \mathbf{t}, \widetilde{\mathbf{w}})| \leq (3\exp(\eta M\bar{t}) + 3)^{j-1} \cdot \epsilon.$$

Then by focusing on interval $[t_j, t_{j+1}]$ and using Lemma G.6 again, one can show that

$$\sup_{t \in [t_j, t_{j-1}]} |\widetilde{\mathbf{x}}^\eta(t, x; \mathbf{t}, \mathbf{w}) - \widetilde{\mathbf{y}}_{\lfloor t \rfloor}^\eta(y; \mathbf{t}, \widetilde{\mathbf{w}})| \leq 2\epsilon + \left((3\exp(\eta M\bar{t}) + 3)^{j-1} + 1\right)\exp(\eta M\bar{t})\epsilon$$

$$\leq (3\exp(\eta M\bar{t}) + 3)^j \cdot \epsilon.$$

This concludes the proof. $\qquad \square$

In the next few results, we show that the same can be said for gradient descent iterates $\widetilde{\mathbf{y}}_n$ and the SGD iterates $X_n$. Specifically, our first goal is to show that before any *large* jump (see the definition in G.21), it is unlikely that the gradient descent process $\mathbf{y}_n^\eta$ would deviate too far from $X_n^\eta$. Define the event

$$A(n, \eta, \epsilon, \delta) = \left\{ \max_{k=1,2,\cdots,n \wedge (T_1^\eta(\delta)-1)} \eta|Z_1 + \cdots + Z_k| \leq \epsilon \right\} \quad \text{(G.42)}$$

and recall that arrival times $T_j^\eta(\delta)$ are defined in eq. (G.22).

As a building block, we first study the case when the starting point $x$ is close to the reflection boundary $-L$. The takeaway from the next result is that the reflection operator hardly comes into play, since the SGD iterates would most likely quickly move to somewhere far enough from $\pm L$; besides, throughout this procedure the SGD iterates would most likely stay pretty close to the corresponding deterministic gradient descent process.

**Lemma G.8.** *Given $\epsilon \in (0, \bar{\epsilon}/9)$, it holds for any sufficiently small $\epsilon, \delta, \eta > 0$ that, if $x \in [-L, -L + \bar{\epsilon}]$ and $\rho_0(|x - y| + 9\epsilon) < \bar{\epsilon}$, then on event $A(n, \eta, \epsilon, \delta)$ we have*

$$|X_k^\eta(x) - \mathbf{y}_k^\eta(y)| \le \rho_0 \cdot (|x - y| + 9\epsilon) \quad \forall k = 1, 2, \cdots, n \wedge (T_1^\eta(\delta) - 1) \wedge \widetilde{T}_{escape}^\eta(x)$$

*where $\widetilde{T}_{escape}^\eta(x) \triangleq \min\{n \ge 0 : X_n^\eta(x) > -L + \bar{\epsilon}\}$ and $\rho_0 \triangleq \exp\left(\frac{2M\bar{\epsilon}}{0.9c_-^L}\right)$ is a constant that does not vary with our choice of $\epsilon, \delta, \eta$.*

*Proof.* For any $k < T_1^\eta(\delta)$, we know that $Z_k = Z_k^{\le\delta}$ (thus $\eta|Z_k| < \delta$). Also, recall that $|f'(x)| \le M$ for any $x \in \Omega_i$. Therefore, as long as $\eta$ and $\delta$ are small enough, we will have that

$$|\eta(-f'(X_n^\eta(x)) + Z_k^{\le\delta})| \le b \tag{G.43}$$

so the gradient clipping operator in eq. (G.12) has no effect when $k < T_1^\eta(\delta)$, and in fact the only possible time for the gradient clipping trick to work is at $T_j^\eta(\delta)$. Therefore, we can safely rewrite the SGD update as

$$X_k^\eta(x) = X_{k-1}^\eta(x) - \eta f'(X_{k-1}^\eta(x)) + \eta Z_k + R_k \quad \forall k < T_1^\eta(\delta)$$

where each $R_k \ge 0$ and it represents the push caused by reflection at $-L$.

First, choose $\epsilon$ small enough so that $9\epsilon < \bar{\epsilon}$. Next, based on eq. (G.31) we have the following lower bound:

$$X_k^\eta(x) \ge x + 0.9c_-^L \eta k - \epsilon \quad \forall k < T_1^\eta(\delta).$$

Let $\widetilde{t}_0(x, \epsilon) \triangleq \min\{n \ge 0 : X_n^\eta(x) \ge -L + 2\epsilon\}$. Due to the inequality above, we know that

$$\widetilde{t}_0(x, \epsilon) \le \frac{3\epsilon}{0.9c_-^L \eta}. \tag{G.44}$$

One the other hand, given the current choice of $\epsilon$, if we choose $\eta$ and $\delta$ small enough, then using the same argument leading to eq. (G.43), we will have

$$X_{\widetilde{t}_0(x,\epsilon)}^\eta(x) \le -L + 2.1\epsilon \le x + 1.1c_-^L \eta k + 2.1\epsilon$$

if $\widetilde{t}_0(x, \epsilon) \ge 1$ (namely $x < -L + 2\epsilon$).

Let us inspect the two scenarios separately. First, assume $\widetilde{t}_0(x, \epsilon) \ge 1$. For the deterministic gradient descent process $\mathbf{y}_n^\eta(y)$, we have the following bounds:

$$y + 0.9c_-^L \eta k \le \mathbf{y}_k^\eta(y) \le y + 1.1c_-^L \eta k \quad \forall k \le \widetilde{t}_0(x, \epsilon) \wedge (T_1^\eta(\delta) - 1).$$

This gives us

$$|X_k^\eta(x) - \mathbf{y}_k^\eta(y)| \le |x - y| + 0.2c_-^L \eta k + 2.1\epsilon \quad \forall k \le \widetilde{t}_0(x, \epsilon) \wedge (T_1^\eta(\delta) - 1).$$

At time $k = \widetilde{t}_0(x, \epsilon)$, due to previous bound on $\widetilde{t}_0(x, \epsilon)$, we know that $|X_{\widetilde{t}_0(x,\epsilon)}^\eta(x) - \mathbf{y}_{\widetilde{t}_0(x,\epsilon)}^\eta(y)| \le |x - y| + 7\epsilon$. If $n \wedge (T_1^\eta(\delta) - 1) \le \widetilde{t}_0(x, \epsilon)$ then we have already shown the desired claim. Otherwise, starting from time $\widetilde{t}_0(x, \epsilon)$, due to the definition of event $A(n, \eta, \epsilon, \delta)$ in eq. (G.42), we know that the SGD iterates $X_n^\eta(x)$ will not touch the boundary $-L$ afterwards. Therefore, by directly applying Lemma G.5, and notice that $|f''(x)| \le M$ for any $x \in [-L, L]$ and, we have

$$|X_k^\eta(x) - \mathbf{y}_k^\eta(y)| \le (|x - y| + 9\epsilon) \cdot \exp\left(\frac{2M\bar{\epsilon}}{0.9c_-^L}\right) \quad \forall k \le n \wedge (T_1^\eta(\delta) - 1) \wedge \widetilde{T}_{escape}^\eta(x).$$

Indeed, it suffices to use Lemma G.5 for the next $\lceil 2\bar{\epsilon}/(0.9\eta c_-^L) \rceil$ steps to show that $|X_k^\eta(x) - \mathbf{y}_k^\eta(y)|$ should be smaller than $\epsilon$ for the next $\lceil 2\bar{\epsilon}/(0.9\eta c_-^L) \rceil$ steps, while $\mathbf{y}_k^\eta(y)$ will reach some where in $(-L + 2\bar{\epsilon}, -L + 3\bar{\epsilon})$ within $\lceil 2\bar{\epsilon}/(0.9\eta c_-^L) \rceil$ steps so we must have

$$n \wedge (T_1^\eta(\delta) - 1) \wedge \widetilde{T}_{\text{escape}}^\eta(x) \wedge \widetilde{t}_0(x, \epsilon) - \widetilde{t}_0(x, \epsilon) \leq 2\bar{\epsilon}/(0.9\eta c_-^L) \tag{G.45}$$

Lastly, in the case $\widetilde{t}_0(x, \epsilon) = 0$ (which means $x \geq -L + \epsilon$), we can use Lemma G.5 directly as we did above and establish the same bound. This concludes the proof. $\square$

Obviously, a similar result can be shown if $x$ is in the rightmost attraction field $(s_{n_{\min-1}}, L]$ and the approach is identical. We omit the details here. In the next Lemma, we consider the scenario where the starting point $x$ is far enough from the boundaries.

**Lemma G.9.** *Given any $\epsilon > 0$, the following holds for all sufficiently small $\eta > 0$: for any $x, y \in \Omega$ and positive integer $n$ such that $|x - L| > 2\epsilon, |x + L| > 2\epsilon, |x - s_-| > 2\epsilon, |x - s_+| > 2\epsilon$ and $|x - y| < \frac{\epsilon}{2\exp(\eta M n)}$, on event*

$$A\left(n, \eta, \frac{\epsilon}{2\exp(\eta M n)}, \delta\right) \cap \left\{ |\mathbf{y}_j^\eta(y)| \in \Omega, |X_j^\eta(x)| \in \Omega \ \forall j = 1, 2, \cdots, n \wedge (T_1^\eta(\delta) - 1) \right\}$$

*we have*

$$|\widetilde{X}_m^\eta(y) - X_m^\eta(x)| \leq \epsilon \ \forall m = 1, 2, \cdots, n \wedge (T_1^\eta(\delta) - 1).$$

*Proof.* For sufficiently small $\eta$, we will have that the (deterministic) gradient descent iterates $|\mathbf{y}_n^\eta|$ is monotonically decreasing in $n$, which ensures that $\mathbf{y}_n^\eta$ always stays in the range that are at least $\epsilon$−away from $\pm L$ or $s_-, s_+$. We now show that the claim holds for any such $\eta$.

On event $\left\{ |\mathbf{y}_j^\eta(y)| \in \Omega, |X_j^\eta(x)| \in \Omega \ \forall j = 1, 2, \cdots, n \wedge (T_1^\eta(\delta) - 1) \right\}$, we are able to apply Lemma G.5 inductively for any $m \in [n]$ and obtain that

$$|\mathbf{y}_j^\eta(y) - X_j^\eta(x)| \leq \left(|x - y| + \frac{\epsilon}{2\exp(\eta M n)}\right) \exp(\eta M j) < \epsilon \ \forall j = 1, 2, \cdots, m$$

and conclude the proof. The reason to apply the Lemma inductively for $m = 1, 2, \cdots, n$, instead of directly at step $n$, is to ensure that SGD iterates $X_n^\eta$ would not hit the boundary $\pm L$ (so the reflection operator would not come into play on the time interval we are currently interested in), thus ensuring that Lemma G.5 is applicable. $\square$

Similar to the extension from Lemma G.6 to Corollary G.7, we can extend Lemma G.9 to show that, if we consider the a gradient descent process that is only perturbed by large noises, then it should stay pretty close to the SGD iterates $X_n^\eta$. To be specific, let

$$Y_0^\eta(x) = x \tag{G.46}$$

$$Y_n^\eta(x) = \varphi_L\left(Y_{n-1}^\eta(x) - \varphi_b\left(-\eta f'\left(Y_{n-1}^\eta(x)\right) + \sum_{j \geq 1} \mathbb{1}\{n = T_j^\eta(\delta)\}\eta Z_n\right)\right). \tag{G.47}$$

be a gradient descent process (with gradient clipping at threshold $b$) that is only shocked by large noises in $(Z_n)_{n \geq 1}$. The next corollary can be shown by an approach that is identical to Corollary G.7 (namely, inductively repeating Lemma G.9 at each jump time) so we omit the details here.

**Corollary G.10.** *Given any $\epsilon > 0$, the following holds for any sufficiently small $\eta > 0$: For any $|x| < 2\epsilon$, on event $A_0(\epsilon, \eta, \delta) \cap B_0(\epsilon, \eta, \delta)$, we have*

$$|Y_n^\eta(x) - X_n^\eta(x)| < \widetilde{\rho}\epsilon \ \forall n = 1, 2, \cdots, T_{l^*}^\eta(\delta)$$

*where*

$$A_0(\epsilon, \eta, \delta) \triangleq \left\{ \forall i = 1, \cdots, l^*, \max_{j = T_{i-1}^\eta(\delta)+1, \cdots, T_i^\eta(\delta)-1} \eta |Z_{T_{i-1}^\eta(\delta)+1} + \cdots + Z_j| \leq \frac{\epsilon}{2\exp(2\bar{t}M)} \right\};$$

$$B_0(\epsilon, \eta, \delta) \triangleq \left\{ \forall j = 2, \cdots, l^*, T_j^\eta(\delta) - T_{j-1}^\eta(\delta) \leq 2\bar{t}/\eta \right\}$$

*and $\widetilde{\rho} \in (0, \infty)$ is a constant that does not vary with $\eta, \delta, \epsilon$.*

The next two results shows that the type of events $A(n, \eta, \epsilon, \delta)$ defined in eq. (G.42) is indeed very likely to occur, especially for small $\epsilon$. For clarity of the presentation, we introduce the following definitions that are slightly more general than the *small* and *large* jumps defined in eq. (G.20)eq. (G.21) (for any $c > 0$)

$$Z_n^{\leq c} \triangleq Z_n \mathbb{1}\{|Z_n| \leq c\},$$
$$Z_n^{> c} \triangleq Z_n \mathbb{1}\{|Z_n| > c\}.$$

**Lemma G.11.** *Define functions* $u(\eta) = \delta/\eta^{1-\Delta}$, $v(\eta) = \epsilon\eta^{\widetilde{\Delta}}$ *with* $\epsilon, \delta > 0$. *If real numbers* $\Delta, \widetilde{\Delta}, \beta, \epsilon, \delta$ *and positive integers* $j, N$ *are such that the following conditions hold:*

$$\Delta \in \big[0, (1 - \frac{1}{\alpha}) \wedge \frac{1}{2}\big), \tag{G.48}$$

$$\beta \in \big(1, (2 - 2\Delta) \wedge \alpha(1 - \Delta)\big), \tag{G.49}$$

$$\widetilde{\Delta} \in [0, \frac{\Delta}{2}], \quad \widetilde{\Delta} < \alpha(1 - \Delta) - \beta, \tag{G.50}$$

$$N < \big(\alpha(1 - \Delta) - \beta\big)j, \tag{G.51}$$

$$v(\eta) - j\eta u(\eta) \geq v(\eta)/2 \quad \text{for all } \eta > 0 \text{ sufficiently small,} \tag{G.52}$$

*then*

$$\mathbb{P}\Big(\max_{k=1,2,\cdots,\lceil 1/\eta^\beta \rceil} \eta\big|Z_1^{\leq u(\eta)} + \cdots + Z_k^{\leq u(\eta)}\big| > 3v(\eta)\Big) = o(\eta^N)$$

*as* $\eta \downarrow 0$.

*Proof.* From the stated range of the parameters, we know that

$$\alpha(1 - \Delta) > \beta,$$
$$\big(\alpha(1 - \Delta) - \beta\big)j > N,$$

so we are able to find $\gamma \in (0, 1)$ small enough such that

$$\alpha(1 - \Delta)(1 - 2\gamma) > \beta, \tag{G.53}$$

$$\big(\alpha(1 - \Delta)(1 - 2\gamma) - \beta\big)j > N. \tag{G.54}$$

Fix such $\gamma \in (0, 1)$ for the rest of the proof, and let $n(\eta) \triangleq \lceil (1/\eta)^\beta \rceil$, $I \triangleq \#\big\{i \in [n(\eta)] : |Z_i^{\leq u(\eta)}| > u(\eta)^{1-\gamma}\big\}$. Then

$$\mathbb{P}\Big(\big|Z_1^{\leq u(\eta)} + \cdots + Z_{n(\eta)}^{\leq u(\eta)}\big| > v(\eta)\Big)$$

$$= \sum_{i=0}^{j-1} \underbrace{\mathbb{P}\Big(\big|Z_1^{\leq u(\eta)} + \cdots + Z_{n(\eta)}^{\leq u(\eta)}\big| > v(\eta), I = i\Big)}_{\triangleq (\mathrm{I})} + \underbrace{\mathbb{P}\Big(\big|Z_1^{\leq u(\eta)} + \cdots + Z_{n(\eta)}^{\leq u(\eta)}\big| > v(\eta), I \geq j\Big)}_{\triangleq (\mathrm{II})}$$

Note that since $|Z_i^{\leq u(\eta)}| < u(\eta)$,

$$(\mathrm{I}) \leq \binom{n(\eta)}{i} \cdot \mathbb{P}\Big(\big|Z_1^{\leq u(\eta)} + \cdots + Z_{n(\eta)-i}^{\leq u(\eta)}\big| > \frac{v(\eta) - i\eta u(\eta)}{\eta}, |Z_i^{\leq u(\eta)}| \leq u(\eta)^{1-\gamma} \, \forall i \in [n(\eta) - i]\Big)$$

$$\leq n(\eta)^i \cdot \mathbb{P}\Big(\big|Z_1^{\leq u(\eta)^{1-\gamma}} + \cdots + Z_{n(\eta)-i}^{\leq u(\eta)^{1-\gamma}}\big| > \frac{v(\eta) - i\eta u(\eta)}{\eta}\Big)$$

$$\leq n(\eta)^i \cdot \mathbb{P}\Big(\big|Z_1^{\leq u(\eta)^{1-\gamma}} + \cdots + Z_{n(\eta)-i}^{\leq u(\eta)^{1-\gamma}}\big| > \frac{v(\eta)}{2\eta}\Big) \tag{G.55}$$

where the last inequality follows from eq. (G.52). First, since $\mathbb{E}Z_1 = 0$, we have

$$\big|\mathbb{E}Z_1^{\leq u(\eta)^{1-\gamma}}\big| = \big|\mathbb{E}Z_1^{> u(\eta)^{1-\gamma}}\big|$$

$$= \int_{u(\eta)^{1-\gamma}}^{\infty} \mathbb{P}(|Z_1| > x)dx \in \mathcal{RV}_{(\alpha-1)(1-\gamma)(1-\Delta)}(\eta).$$

Therefore, for all $\eta > 0$ that are sufficiently small,

$$|\mathbb{E}Z_1^{\leq u(\eta)^{1-\gamma}} + \cdots + \mathbb{E}Z_{n(\eta)-i}^{\leq u(\eta)^{1-\gamma}}|$$
$$\leq n(\eta) \cdot \eta^{(\alpha-1)(1-\Delta)(1-2\gamma)} \leq 2\eta^{(\alpha-1)(1-\Delta)(1-2\gamma)-\beta}$$
$$\leq (1/\eta)^{(1-\Delta)(1-2\gamma)} \quad \text{due to eq. (G.53)}$$
$$\leq \frac{v(\eta)}{4\eta} \quad \text{due to } \widetilde{\Delta}/2 \leq \Delta \text{ in eq. (G.50) and } 1 - \gamma < 1.$$

If we let $Y_n = Z_n^{\leq u(\eta)^{1-\gamma}} - \mathbb{E}Z_n^{\leq u(\eta)^{1-\gamma}}$ and plug the bound above back into eq. (G.55), then (for all $\eta > 0$ that are sufficiently small)

$$(\text{I}) \leq n(\eta)^i \cdot \mathbb{P}(|Y_1 + \cdots + Y_{n(\eta)-i}| > \frac{v(\eta)}{4\eta})$$
$$\leq n(\eta)^i \exp\left(-\frac{\frac{\epsilon^2}{16} \cdot 1/\eta^{2-2\widetilde{\Delta}}}{2(n(\eta)-i)\mathbb{E}|Y_1|^2 + \frac{2}{3}\delta^{1-\gamma} \cdot (1/\eta)^{(1-\Delta)(1-\gamma)} \cdot \frac{\epsilon}{4}/\eta^{1-\widetilde{\Delta}}}\right) \quad \text{(G.56)}$$

where the last inequality is obtained from Bernstein's inequality. Note that from Karamata's theorem,

$$\mathbb{E}|Y_1|^2 = var(Z_1^{\leq u(\eta)^{1-\gamma}}) \leq \mathbb{E}|Z_1^{\leq u(\eta)^{1-\gamma}}|^2$$
$$\leq \int_0^{u(\eta)^{1-\gamma}} 2x\mathbb{P}(|Z_1| > x)dx \in \mathcal{RV}_{-(1-\Delta)(1-\gamma)(2-\alpha)}(\eta).$$

Now note that

- In case that $\alpha < 2$, for all $\eta > 0$ that are sufficiently small, we have (using eq. (G.50))
$$2(n(\eta)-i)\mathbb{E}|Y_1|^2 \leq (1/\eta)^{\beta+(2-\alpha)(1-\Delta)} < (1/\eta)^{2(1-\Delta)}$$
$$\Rightarrow \frac{1/\eta^{2-2\widetilde{\Delta}}}{2(n(\eta)-i)\mathbb{E}|Y_1|^2} \geq 1/\eta^\Delta;$$

- In case that $\alpha \geq 2$, for all $\eta > 0$ that are sufficiently small,
$$2(n(\eta)-i)\mathbb{E}|Y_1|^2 < 1/\eta^{\beta+\frac{\Delta}{2}}$$
and we know that $\beta + \frac{\Delta}{2} < 2 - 2\widetilde{\Delta}$ due to $2 - \beta > 2\Delta$ and $2\widetilde{\Delta} \leq \Delta$ (see eq. (G.48)-eq. (G.50));

- Since $\gamma > 0$ and $2\widetilde{\Delta} \leq \Delta$, we know that
$$(1-\Delta)(1-\gamma) + (1-\widetilde{\Delta}) < 2 - 2\widetilde{\Delta}.$$

Therefore, it is easy to see that the R.H.S. of eq. (G.56) decays at a geometric rate as $\eta$ tends to zero, hence $o(\eta^N)$. On the other hand,

$$(\text{II}) \leq \mathbb{P}(I \geq j) \leq \binom{n(\eta)}{j} \cdot \mathbb{P}\left(|Z_i^{\leq u(\eta)}| > u(\eta)^{1-\gamma} \; \forall i = 1, \ldots, j\right)$$
$$\leq n(\eta)^j \cdot \mathbb{P}\left(|Z_1^{\leq u(\eta)}| > u(\eta)^{1-\gamma}\right)^j,$$

which is regularly varying w.r.t. $\eta$ with index $(\alpha(1-\gamma)(1-\Delta) - \beta)j$. Therefore, for all $\eta > 0$ sufficiently small,

$$(\text{II}) \leq \eta^{(\alpha(1-2\gamma)(1-\Delta)-\beta)j} < \eta^N \quad \text{due to eq. (G.54).}$$

Collecting results above, we have established that

$$\mathbb{P}\left(\eta|Z_1^{\leq u(\eta)} + \cdots + Z_{n(\eta)}^{\leq u(\eta)}| > v(\eta)\right) = o(\eta^N).$$

The conclusion of the lemma now follows from Etemadi's theorem. $\square$

Now consider the following setting. Let us fix some positive integer $N$ and $\beta \in (1, 2 \wedge \alpha)$. Then we can find some positive integer $j$ such that $(\alpha - \beta)j > N$. Meanwhile, given any $\epsilon > 0$, we will have $\epsilon - j\delta \geq \epsilon/2$ for all $\delta > 0$ sufficiently small. Therefore, by applying Lemma G.11 with $\Delta = \widetilde{\Delta} = 0$ (hence $u(\eta) = \delta/\eta$, $v(\eta) = \epsilon$) and $\beta, j, N, \epsilon, \delta$ as described here, we immediately get the following result.

**Lemma G.12.** *Given any $\beta \in (1, \alpha \wedge 2)$, $\epsilon > 0$, and $N > 0$, the following holds for any sufficiently small $\delta > 0$:*

$$\mathbb{P}\Big( \max_{j=1,2,\cdots,\lceil (1/\eta)^\beta \rceil} \eta |Z_1^{\leq \delta/\eta} + \cdots + Z_j^{\leq \delta/\eta}| > \epsilon \Big) = o(\eta^N)$$

*as $\eta \downarrow 0$.*

Using results and arguments above, we are able to illustrate the typical behavior of the SGD iterates $X_n^\eta$ in the following two scenarios. First, we show that, when starting from most parts in the attraction field $\Omega$, the SGD iterates $X_n^\eta$ will most likely return to the neighborhood of the local minimum within a short period of time without exiting $\Omega$. Given that there are only finitely many attraction fields on $f$, it is easy to see that the key technical tool Lemma G.1 follows immediately from the next result.

**Lemma G.13.** *For sufficiently small $\epsilon > 0$, the following claim holds:*

$$\lim_{\eta \downarrow 0} \sup_{x \in \Omega : |x-s_-| \wedge |x-s_+| > \epsilon} \mathbb{P}_x \Big( X_n^\eta \in \Omega \ \forall n \leq T_{return}(\eta, \epsilon), and \ T_{\text{return}}(\eta, \epsilon) \leq \rho(\epsilon)/\eta \Big) = 1$$

*where the stopping time involved is defined as*

$$T_{return}(\eta, \epsilon) \triangleq \min\{n \geq 0 : X_n^\eta(x) \in [-2\epsilon, 2\epsilon]\}$$

*the function $\hat{t}(\epsilon)$ is defined in eq. (G.38), and the function $\rho(\cdot)$ is defined as $\rho(\epsilon) = \frac{3\bar{\epsilon}}{0.9 c_-^L \wedge c_+^L} + 2\hat{t}(\epsilon)$*

*Proof.* Throughout, we only consider $\epsilon$ small enough so that Lemma G.8 could hold. Also, fix some $N > 0, \Delta_\alpha \in (0, \alpha - 1)$ and $\beta \in (1, \alpha)$. Let $\sigma(x, \eta) \triangleq \min\{n \geq 0 : X_n^\eta \notin \Omega\}$.

Without loss of generality, we assume $\Omega = [-L, s_+)$ and $x < 0$ (so reflection at $-L$) is a possibility. Any other case can be addressed similarly as shown below.

From Lemma G.4 and the regular varying nature of $H(\cdot)$, we have, for any $\epsilon, \delta > 0$,

$$\mathbb{P}(T_1^\eta(\delta) \leq \rho(\epsilon)/\eta) \leq \eta^{\alpha - 1 - \Delta_\alpha} \tag{G.57}$$

for any sufficiently small $\eta$.

Let $\widetilde{T}_{\text{escape}}^\eta(x)$ be the stopping time defined in Lemma G.8. From eq. (G.44),eq. (G.45),eq. (G.57) and Lemma G.12, we know that

$$\sup_{x \in [-L, -L+\bar{\epsilon}]} \mathbb{P}\Big( \widetilde{T}_{\text{escape}}^\eta(x) < \sigma(x, \eta), \widetilde{T}_{\text{escape}}^\eta(x) \leq \frac{3\bar{\epsilon}}{0.9 c_-^L \eta} \text{ and } X_{\widetilde{T}_{\text{escape}}^\eta}^\eta(x) \in [-L+\bar{\epsilon}, -L+2\bar{\epsilon}] \Big)$$
$$\geq 1 - \eta^N - \eta^{\alpha - 1 - \Delta_\alpha} \tag{G.58}$$

for all sufficiently small $\eta$.

Next, we focus on $x \in \Omega$ such that $|x - s_-| \wedge |x - s_+| > \epsilon$ and $x \geq -L + \bar{\epsilon}$. We start by considering the time it took for the (deterministic) gradient descent process $\mathbf{y}_n^\eta(x)$ to return to $[-1.5\epsilon, 1.5\epsilon]$. From the definition of $\hat{t}(\epsilon)$ in eq. (G.38) and Lemma G.6, we know that for $\eta$ small enough such that $\eta \exp(2M\hat{t}(\epsilon)) < 0.5\epsilon$, we have

$$\min\{n \geq 0 : \mathbf{y}_n^\eta(x) \in [-1.5\epsilon, 1.5\epsilon]\} \leq 2\hat{t}(\epsilon)/\eta.$$

Now consider event $A(\lceil (1/\eta)^\beta \rceil, \eta, \frac{\epsilon}{4 \exp(2M\hat{t}(\epsilon))}, \delta)$ (see definition in eq. (G.42)). From Lemma G.12, we know that for any sufficiently small $\delta$, we have

$$\mathbb{P}\Big( \big( A(\lceil (1/\eta)^\beta \rceil, \eta, \frac{\epsilon}{4 \exp(2M\hat{t}(\epsilon))}, \delta) \big)^c \Big) = o(\eta^N). \tag{G.59}$$

Combining this result with eq. (G.57)eq. (G.59) and Lemma G.9, we get

$$\sup_{x\in\Omega:|x-s_-|\wedge|x-s_+|>\epsilon,x\geq-L+\bar{\epsilon}} \mathbb{P}_x\Big(T_{\text{return}}(\eta,\epsilon) < \sigma(x,\eta), T_{\text{return}}(\eta,\epsilon) \leq 2\hat{t}(\epsilon)/\eta\Big) \tag{G.60}$$

$$\geq 1 - \eta^N - \eta^{\alpha-1-\Delta_\alpha} \tag{G.61}$$

for any sufficiently small $\eta$. To conclude the proof, we only to combine strong Markov property (at $\widetilde{T}^\eta_{\text{escape}}$) with bounds in eq. (G.58)eq. (G.61). $\qquad\square$

In the next result, we show that, once entering a $\epsilon-$small neighborhood of the local minimum, the SGD iterates will most likely stay there until the next large jump.

**Lemma G.14.** *Given $N_0 > 0$, the following claim holds for any $\epsilon, \delta > 0$ that are sufficiently small:*

$$\sup_{x\in[-2\epsilon,2\epsilon]} \mathbb{P}\Big(\exists n < T^\eta_1(\delta) \ s.t. \ |X^\eta_n(x)| > 3\epsilon\Big) = o(\eta^{N_0})$$

*as $\eta \downarrow 0$.*

*Proof.* Fix $\epsilon$ small enough such that $3\epsilon < \epsilon_0$ (see Assumption 1 for the constant $\epsilon_0$). Also, fix some $\Delta\alpha \in (0,1), \beta \in (1,\alpha), N > \alpha + \Delta\alpha - \beta + N_0$. Due to Lemma G.12, for any $\delta$ sufficiently small, we will have

$$\mathbb{P}\Big(\max_{j=1,2,\cdots,\lceil(1/\eta)^\beta\rceil} \eta|Z^{\leq,\delta}_1 + \cdots + Z^{\leq,\delta}_j| > \frac{\epsilon}{\exp(2M)}\Big) = o(\eta^N). \tag{G.62}$$

Fix such $\delta > 0$. We now show that the desired claim is true for the chosen $\epsilon, \delta$.

First of all, from Lemma G.3, we know the existence of some $\theta > 0$ such that

$$\mathbb{P}(T^\eta_1(\delta) > 1/\eta^{\alpha+\Delta\alpha}) = o(\exp(-1/\eta^\theta)). \tag{G.63}$$

Next, let us zoom in on the first $\lceil(1/\eta)^\beta\rceil$ SGD iterates. For any $\eta$ small enough, we will have $\mathbf{y}^\eta_n(x) \in [-2\epsilon,2\epsilon]$ for any $n \geq 1$ and $\mathbf{y}^\eta_{\lceil(1/\eta)^\beta\rceil}(x) \in [-\epsilon,\epsilon]$ given $x \in [-2\epsilon,2\epsilon]$. From now on we only consider such $\eta$. Due to Lemma G.9, we know that on event $\big\{\max_{j=1,2,\cdots,\lceil(1/\eta)\rceil} \eta|Z^{\leq,\delta}_1 + \cdots + Z^{\leq,\delta}_j| > \frac{\epsilon}{\exp(2M)}\big\}$, we have

$$|X^\eta_n(x)| \leq 3\epsilon \ \ \forall n \leq \lceil 1/\eta^\beta\rceil \wedge (T^\eta_1(\delta) - 1)$$

and on event $\big\{\max_{j=1,2,\cdots,\lceil(1/\eta)\rceil} \eta|Z^{\leq,\delta}_1 + \cdots + Z^{\leq,\delta}_j| > \frac{\epsilon}{\exp(2M)}\big\} \cap \{T^\eta_1(\delta) > \lceil(1/\eta)^\beta\rceil\}$, we have $X^\eta_{T^\eta_1(\delta)}(x) \in [-2\epsilon,2\epsilon]$. Now by repeating the same argument inductively for $\lceil 1/\eta^{\alpha+\Delta\alpha-\beta}\rceil$ times, we can show that on event

$$\{\forall i = 1,2,\cdots,\lceil\frac{1}{\eta^{\alpha+\Delta\alpha-\beta}}\rceil, \max_{j=1,\cdots,\lceil(1/\eta)^\beta\rceil} \eta|Z^{\leq,\delta}_{i\lceil(1/\eta)^\beta\rceil+1} + \cdots + Z^{\leq,\delta}_{i\lceil(1/\eta)^\beta\rceil+j}| > \frac{\epsilon}{\exp(2M)}\},$$

we have $|X^\eta_n(x)| \leq 3\epsilon \ \forall n \leq 1/\eta^{\alpha+\Delta\alpha} \wedge (T^\eta_1(\delta) - 1)$. To conclude the proof, we only need to combine this fact with eq. (G.62). $\qquad\square$

We introduce a few concepts that will be crucial in the analysis below. Recall the definition of perturbed ODE $\widetilde{\mathbf{x}}^\eta$ in eq. (G.32)-eq. (G.34) (note that we will drop the notational dependency on learning rate $\eta$ when we choose $\eta = 1$). Consider the definition of the following two mappings from where $\mathbf{w} = (w_1,\cdots,w_{l^*})$ is a sequence of real numbers and $\mathbf{t} = (t_1,t_2,\cdots,t_{l^*})$ with $0 = t_1 < t_2 < t_3 < \cdots$ as

$$h(\mathbf{w},\mathbf{t}) = \widetilde{\mathbf{x}}(t_{l^*},0;\mathbf{t},\mathbf{w}).$$

Next, define sets (for any $\epsilon \in (-\bar{\epsilon},\bar{\epsilon})$)

$$E(\epsilon) = \{(\mathbf{w},\mathbf{t}) \subseteq \mathbb{R}^{l^*} \times \Big(\mathbb{R}_+\Big)^{l^*-1} : h(\mathbf{w},\mathbf{t}) \notin [s_- - \epsilon, s_+ + \epsilon]\}. \tag{G.64}$$

We add a few remarks about the two types of sets defined above.

- Intuitively speaking, $E(\epsilon)$ contains all the perturbations (with times and sizes) that can send the ODE out of the current attraction field (allowing for some error with size $\epsilon$);
- From the definition of $\bar{t}, \bar{\delta}$ in eq. (G.35)eq. (G.36) and Corollary G.7, one can easily see that for a fixed $\epsilon \in (-\bar{\epsilon}, \bar{\epsilon})$,

$$(\mathbf{w}, \mathbf{t}) \in E(\epsilon) \Rightarrow |w_j| > \bar{\delta}, t_j - t_{j-1} \leq \bar{t} \; \forall j;$$

- Lastly, $E(\epsilon)$ are open sets due to $f \in C^2$.

Use $\mathbf{Leb}_+$ to denote the Lebesgue measure restricted on $[0, \infty)$, and define (Borel) measure $\nu_\alpha$ with density on $\mathbb{R} \setminus \{0\}$:

$$\nu_\alpha(dx) = \mathbb{1}\{x > 0\}\frac{\alpha p_+}{x^{\alpha+1}} + \mathbb{1}\{x < 0\}\frac{\alpha p_-}{|x|^{\alpha+1}}$$

where $\alpha > 1$ is the regular variation index for the distribution of $Z_1$ and $p_-, p_+ \in (0, 1)$ are constants in Assumption 2. Now we can define a Borel measure $\mu$ on $\mathbb{R}^{l^*} \times \left(\mathbb{R}_+\right)^{l^*-1}$ as product measure

$$\mu = (\nu_\alpha)^{l^*} \times (\mathbf{Leb}_+)^{l^*-1}. \tag{G.65}$$

Due to remarks above, one can see that for $\epsilon \in (-\bar{\epsilon}, \bar{\epsilon})$, we have $\mu(E(\epsilon)) < \infty$. We are now ready to analyze a specific type of noise $Z_n$.

**Definition G.1.** *For any $n \geq 1$ and any $\epsilon \in (-\bar{\epsilon}, \bar{\epsilon}), \delta \in (0, b \wedge \bar{\delta}), \eta > 0$, we say that the jump $Z_n$ has $(\epsilon, \delta, \eta)$-**overflow** if*

- $\eta|Z_n| > \delta$;
- *In the set $\{n + 1, \cdots, n + 2\lceil l^*\bar{t}/\eta \rceil\}$, there are at least $(l^* - 1)$ elements (ordered as $n < t_2 < t_3 < \cdots < t_{l^*}$) such that $\eta|Z_{t_i}| > \delta$ for any $i = 2, \cdots, l^*$;*
- *Let $t_1 = n$ and $t'_i = t_i - t_{i-1}$ for any $i = 2, \cdots, l^*$, $w_i = \eta Z_i$ for any $i = 1, \cdots, l^*$, for real sequence $\mathbf{w} = (w_1, w_2, \cdots, w_{l^*})$ and a sequence of positive number $\mathbf{t} = (\eta(t_i - n))_{i=2}^{l^*}$, we have*

$$(\mathbf{w}, \mathbf{t}) \in E(\epsilon).$$

*Moreover, if $Z_n$ has $(\epsilon, \delta, \eta)-$overflow, then we call $h(\mathbf{w}, \mathbf{t})$ as its $(\epsilon, \delta, \eta)-$**overflow endpoint**.*

Due to the iid nature of $(Z_j)_{j \geq 1}$, let us consider an iid sequence $(V_j)_{j \geq 0}$ where the sequence has the same law of $Z_1$. Note that for any fixed $n \geq 1$, the probability that $Z_n$ has $(\epsilon, \delta, \eta)$-overflow is equal to the probability that $V_0$ has $(\epsilon, \delta, \eta)$-overflow. More specifically, we know that $\mathbb{P}(\eta|V_0| > \delta) = H(\delta/\eta)$, and now we focus on conditional probability admitting the following form:

$$p(\epsilon, \delta, \eta) = \mathbb{P}\Big(V_0 \text{ has } (\epsilon, \delta, \eta)\text{-overflow} \ \Big| \ \eta|V_0| > \delta\Big). \tag{G.66}$$

For any open interval $A = (a_1, a_2)$ such that $A \cap [s_- + \bar{\epsilon}, s_+ - \bar{\epsilon}] = \emptyset$, we also define

$$p(\epsilon, \delta, \eta; A) = \mathbb{P}\Big(V_0 \text{ has } (\epsilon, \delta, \eta)\text{-overflow and the endpoint is in } A \ \Big| \ \eta|V_0| > \delta\Big). \tag{G.67}$$

**Lemma G.15.** *For any $\epsilon \in (-\bar{\epsilon}, \bar{\epsilon}), \delta \in (0, b \wedge \bar{\delta})$, and any open interval $A = (a_1, a_2)$ such that $|a_1| \wedge |a_2| > r - \bar{\epsilon}$ and $|a_1| \neq L, |a_2| \neq L$, we have*

$$\lim_{\eta \downarrow 0} \frac{p(\epsilon, \delta, \eta; A)}{\delta^\alpha \left(\frac{H(1/\eta)}{\eta}\right)^{l^*-1}} = \mu\big(E(\epsilon) \cap h^{-1}(A)\big)$$

*where $\mu$ is the measure defined in eq. (G.65), and $p(\cdot, \cdot, \cdot; A)$ is the conditional probability defined in eq. (G.67).*

*Proof.* Let us start by fixing some notations. Let $T_1 = 0$, and define stopping times $T_j = \min\{n > T_{j-1} : \eta|V_n| > \delta\}$ and inter-arrival times $T'_j = T_j - T_{j-1}$ for any $j \geq 1$, and large jump $W_j = V_{T_j}$ for any $j \geq 0$. Note that: first of all, the pair $(T'_i, W_i)$ is independent of $(T'_j, W_j)$ whenever $i \neq j$; besides, $W_j$ and $T'_j$ are independent for all $j \geq 1$.

Define the following sequence (of random elements) $\mathbf{w} = (w_1, \cdots, w_{l^*})$ and $\mathbf{t} = (t_1, \cdots, t_{l^*})$ by

$$w_j = \eta W_j, \quad t_j = \eta T_j.$$

If $V_0$ has $(\epsilon, \delta, \eta)$-overflow, then the following two events must occur:

- $T'_j \le 2\bar{t}/\eta$ for any $j = 2, \cdots, l^*$;

- $\eta |W_j| > \bar{\delta}$ for any $j = 1, 2, \cdots, l^*$;

- $(\mathbf{w}, \mathbf{t}) \in E(\epsilon)$

Therefore, for sufficiently small $\eta$, we now have

$$
\begin{aligned}
&p(\epsilon, \delta, \eta) \\
&= \Big( \mathbb{P}(T'_1 \le 2\bar{t}/\eta) \Big)^{l^*-1} \cdot \int \mathbb{1}\Big\{ (\mathbf{w}, \mathbf{t}) \in E(\epsilon) \Big\} \\
&\qquad\qquad \cdot \mathbb{P}(\eta W_1 = dw_1 | \eta |W_1| > \delta) \cdots \mathbb{P}(\eta W_{l^*} = dw_{l^*} | \eta |W_{l^*}| > \delta) \\
&\qquad\qquad \cdot \mathbb{P}(\eta T'_2 = dt'_2 | \eta T'_2 \le 2\bar{t}) \cdots \mathbb{P}(\eta T'_{l^*} = dt'_{l^*} | \eta T'_{l^*} \le 2\bar{t}) \\
&= \Big( \mathbb{P}(T'_1 \le 2\bar{t}/\eta) \Big)^{l^*-1} \cdot \mathbb{Q}_{\eta,\delta}\big( E(\epsilon) \cap h^{-1}(A) \big)
\end{aligned}
\tag{G.68}
$$

where $\mathbb{Q}_{\eta,\delta}$ is the Borel-measurable probability measure on $\mathbb{R}^{l^*} \times \big( \mathbb{R}_+ \big)^{l^*-1}$ induced by a sequence of independent random variables $(W_1^\uparrow(\eta, \delta), \cdots, W_{l^*}^\uparrow(\eta, \delta), T_2^\uparrow(\eta, \delta), \cdots, T_{l^*}^\uparrow(\eta, \delta))$ such that

- For any $i = 1, \cdots, l^*$, the distribution of $W_i^\uparrow(\eta, \delta)$ follows from $\mathbb{P}\Big( \eta W_1 \in \cdot \,\Big|\, \eta |W_{l^*}| > \delta \Big)$;

- For any $i = 2, \cdots, l^*$, the distribution of $T_i^\uparrow(\eta, \delta)$ follows from $\mathbb{P}\Big( \eta T_1 \in \cdot \,\Big|\, \eta T_1 \le 2\bar{t} \Big)$;

- $\mathbb{Q}_{\eta,\delta}(\cdot) = \mathbb{P}\Big( (\eta W_1^\uparrow(\eta, \delta), \cdots, \eta W_{l^*}^\uparrow(\eta, \delta), \eta T_2^\uparrow(\eta, \delta), \cdots, \eta \sum_{j=2}^{l^*} T_j^\uparrow(\eta, \delta)) \in \cdot \Big)$.

Now we study the weak convergence of $W_1^\uparrow, T_1^\uparrow$:

- Due to the regularly varying nature of distribution of $Z_1$ (hence for $W_1$), we know that: for any $x > \delta$,

$$\lim_{\eta \downarrow 0} \mathbb{P}\Big( \eta W_1 > x \,\Big|\, \eta |W_{l^*}| > \delta \Big) = p_+ \frac{\delta^\alpha}{x^\alpha}, \quad \lim_{\eta \downarrow 0} \mathbb{P}\Big( \eta W_1 < -x \,\Big|\, \eta |W_{l^*}| > \delta \Big) = p_- \frac{\delta^\alpha}{x^\alpha};$$

therefore, $W_1^\uparrow(\eta, \delta)$ weakly converges to a (randomly signed) Pareto RV that admits the density

$$\nu_{\alpha,\delta}(dx) = \mathbb{1}\{x > 0\} p_+ \frac{\alpha \delta^\alpha}{x^{\alpha+1}} + \mathbb{1}\{x < 0\} p_- \frac{\alpha \delta^\alpha}{|x|^{\alpha+1}}$$

as $\eta \downarrow 0$;

- For any $x \in [0, 2\bar{t}]$, since $\lim_{\eta \downarrow 0} \lfloor x/\eta \rfloor H(\delta/\eta) = 0$, it is easy to show that

$$\lim_{\eta \downarrow 0} \frac{1 - (1 - H(\delta/\eta))^{\lfloor x/\eta \rfloor}}{\lfloor x/\eta \rfloor H(\delta/\eta)} = 1;$$

therefore, we have (for any $x \in (0, 2\bar{t}]$)

$$\mathbb{P}(\eta T_1 \le x \mid \eta T_1 \le 2\bar{t}) = \frac{1 - (1 - H(\delta/\eta))^{\lfloor x/\eta \rfloor}}{1 - (1 - H(\delta/\eta))^{\lfloor 2\bar{t}/\eta \rfloor}} \to \frac{x}{2\bar{t}}$$

as $\eta \downarrow 0$, which implies that $T_1^\uparrow$ converges weakly to a uniform RV on $[0, 2\bar{t}]$.

Let us denote the weak limit of measure $\mathbb{Q}_{\eta,\delta}$ as $\mu_{\delta,2\bar{t}}$. In the discussion before the Lemma we have shown that, for any $(\mathbf{w}, \mathbf{t}) \in E(\epsilon)$ (with $\delta \in (0, \bar{\delta})$), we have $|w_i| \geq \bar{\delta}$ and $|t_i'| \leq 2\bar{t}$; since we require $\delta < \bar{\delta}$, by definition of measures $\mu$ and $\mu_{\delta,2\bar{t}}$ we have

$$\mu_{\delta,2\bar{t}}\big(E(\epsilon) \cap h^{-1}(A)\big) = \frac{\delta^{\alpha l^*}}{(2\bar{t})^{l^*-1}} \cdot \mu\big(E(\epsilon) \cap h^{-1}(A)\big).$$

For simplicity of notations, we let $E(\epsilon, A) \triangleq E(\epsilon) \cap h^{-1}(A)$. By definition of the set $E(\epsilon)$, we have (recall that $A$ is an open interval $(a_1, a_2)$ that does not overlap with $[s_- + \bar{\epsilon}, s_+ - \bar{\epsilon}]$)

$$\begin{aligned}
E(\epsilon, A) &= h^{-1}\big((-\infty, s_- - \epsilon) \cup (s_+ + \epsilon, \infty)\big) \cap h^{-1}\big((a_1, a_2)\big) \\
&= h^{-1}\Big(\big((-\infty, s_- - \epsilon) \cup (s_+ + \epsilon, \infty)\big) \cap (a_1, a_2)\Big) \\
&= h^{-1}\big(F(\epsilon, a_1, a_2)\big)
\end{aligned}$$

where $F(\epsilon, a_1, a_2) \triangleq \big((-\infty, s_- - \epsilon) \cup (s_+ + \epsilon, \infty)\big) \cap (a_1, a_2)$. Meanwhile, it is easy to see that $h$ is a continuous mapping, hence

$$(\mathbf{w}, \mathbf{t}) \in \partial E(\epsilon, A) \Rightarrow h(\mathbf{w}, \mathbf{t}) \in \{s_- + \epsilon, s_+ - \epsilon, a_1, a_2\}.$$

Fix some $s$ with $s \neq \pm L$, $|s| > (l^* - 1)b + \bar{\epsilon}$. For any fixed real numbers $t_2, \cdots, t_{l^*-1}, w_1, \cdots, w_{l^*}$, if $h(w_1, \cdots, w_{l^*}, t_2, \cdots, t_{l^*-1}, t) = s$, then since $\widetilde{\mathbf{x}}(t_{l^*} - 1, 0; w_1, \cdots, w_{l^*}, t_2, \cdots, t_{l^*-1}, t) \in [s - b, s + b]$, due to Assumption 1 (in particular, there is no point $x$ on this interval with $|f'(x)| \leq c_0$), there exists at most one possible $t$ that makes $h(w_1, \cdots, w_{l^*}, t_2, \cdots, t_{l^*-1}, t) = s$. Therefore, let $W_j^*$ be iid RVs from law $\nu_{\alpha,\delta}$ defined above, and $(T_j^{*,\prime})_{j\geq 2}$ be iid RVs from $\mathrm{Unif}[0, \bar{2t}]$, $T_0^* = 0$, $T_k^* = \sum_{j=2}^{k} T_j^{*,\prime}$. By conditioning on all $W_j^*$ and all $T_2^{*,\prime}, \cdots, T_{l^*-1}^{*,\prime}$, we must have

$$\begin{aligned}
\mathbb{P}\Big(h(W_1^*, \cdots, W_j^*, T_2^*, \cdots, T_{l^*}^*) = s \;\Big|\; W_1^* = dw_1, \cdots, W_{l^*}^* = dw_{l^*}, \\
T_2^{*,\prime} = dt_2, \cdots, T_{l^*-1}^{*,\prime} = dt_{l^*-1}\Big) = 0 \qquad \text{(G.69)}
\end{aligned}$$

which implies

$$\mathbb{P}\Big(h(W_1^*, \cdots, W_j^*, T_2^*, \cdots, T_{l^*}^*) = s\Big) = 0$$

hence

$$\mu\Big(\partial E(\epsilon, A)\Big) = 0.$$

By Portmanteau theorem (see Theorem 2.1 of Billingsley (2013)) we have

$$\lim_{\eta\downarrow 0} \mathbb{Q}_{\eta,\delta}(E(\epsilon, A)) = \mu_{\delta,2\bar{t}}(E(\epsilon, A)).$$

Collecting the results we have and using eq. (G.68), we can see that

$$\begin{aligned}
&\limsup_{\eta\downarrow 0} \frac{p(\epsilon, \delta, \eta; A)}{\left(\frac{H(1/\eta)}{\eta}\right)^{l^*-1} \delta^\alpha} \\
&= \limsup_{\eta\downarrow 0} \frac{(2\bar{t})^{l^*-1} \cdot p(\epsilon, \delta, \eta; A)}{\delta^{\alpha l^*} \cdot \left(\mathbb{P}(T_1' \leq 2\bar{t}/\eta)\right)^{l^*-1}} \cdot \left(\frac{\delta^\alpha}{2\bar{t}} \cdot \frac{\mathbb{P}(T_1' \leq 2\bar{t}/\eta)}{H(1/\eta)/\eta}\right)^{l^*-1} \\
&\leq \limsup_{\eta\downarrow 0} \frac{(2\bar{t})^{l^*-1} \cdot p(\epsilon, \delta, \eta; A)}{\delta^{\alpha l^*} \cdot \left(\mathbb{P}(T_1' \leq 2\bar{t}/\eta)\right)^{l^*-1}} \cdot \limsup_{\eta\downarrow 0} \left(\frac{\delta^\alpha}{2\bar{t}} \cdot \frac{\mathbb{P}(T_1' \leq 2\bar{t}/\eta)}{H(1/\eta)/\eta}\right)^{l^*-1} \\
&\leq \mu(E(\epsilon, A)) \cdot \limsup_{\eta\downarrow 0} \left(\frac{\delta^\alpha}{2\bar{t}} \cdot \frac{\mathbb{P}(T_1' \leq 2\bar{t}/\eta)}{H(1/\eta)/\eta}\right)^{l^*-1}.
\end{aligned}$$

Fix some $\kappa > 1$. From Lemma G.4 and the regularly varying nature of function $H$, we get

$$\limsup_{\eta\downarrow 0} \left(\frac{\delta^\alpha}{2\bar{t}} \cdot \frac{\mathbb{P}(T_1' \leq 2\bar{t}/\eta)}{H(1/\eta)/\eta}\right)^{l^*-1} \leq \kappa^{l^*-1} \limsup_{\eta\downarrow 0} \left(\frac{\delta^\alpha}{2\bar{t}} \cdot \frac{2\bar{t}H(\delta/\eta)/\eta}{H(1/\eta)/\eta}\right)^{l^*-1} = \kappa^{l^*-1}.$$

Due to the arbitrariness of $\kappa > 1$, we have established that

$$\limsup_{\eta \downarrow 0} \frac{p(\epsilon, \delta, \eta; A)}{\left( \frac{H(1/\eta)}{\eta} \right)^{l^*-1} \delta^\alpha} \leq \mu(E(\epsilon)).$$

The lower bound can be shown by an argument symmetric to the one for upper bound. $\qquad\square$

The following result is an immediate corollary of Lemma G.15.

**Corollary G.16.** *For any $\epsilon \in (-\bar{\epsilon}, \bar{\epsilon}), \delta \in (0, b \wedge \bar{\delta})$, we have*

$$\lim_{\eta \downarrow 0} \frac{p(\epsilon, \delta, \eta)}{\delta^\alpha \left( \frac{H(1/\eta)}{\eta} \right)^{l^*-1}} = \mu\big(E(\epsilon)\big)$$

*where $\mu$ is the measure defined in eq. (G.65), and $p(\cdot, \cdot, \cdot)$ is the conditional probability defined in eq. (G.66).*

Define the following stopping times:

$$\sigma(\eta) = \min\{n \geq 0 : X_n^\eta \notin \Omega\}; \tag{G.70}$$

$$R(\epsilon, \delta, \eta) = \min\{n \geq T_1^\eta(\delta) : X_n^\eta \in [-2\epsilon, 2\epsilon]\}. \tag{G.71}$$

$\sigma$ indicate the time that the iterates escape the current attraction field, while $R$ denotes the time the SGD iterates return to a small neighborhood of the local minimum after first exit from this small neighborhood. In the next few results, we study the probability of several atypical scenarios when SGD iterates make attempts to escape $\Omega$ or return to local minimum after the attempt fails. First, we show that, when starting from the local minimum, it is very unlikely to escape with less than $l^*$ big jumps.

**Lemma G.17.** *Given $\epsilon \in (0, \bar{\epsilon}), N > 0$, the following claim holds for any sufficiently small $\delta > 0$:*

$$\sup_{x \in [-2\epsilon, 2\epsilon]} \mathbb{P}_x\Big(\sigma(\eta) < R(\epsilon, \eta), \ \sigma(\eta) < T_{l^*}^\eta(\delta)\Big) = o(\eta^N)$$

*as $\eta \downarrow 0$.*

*Proof.* Based on the given $\epsilon > 0$, fix some $\widetilde{\epsilon} = \frac{\epsilon}{4 \exp(2M\hat{t}(\epsilon))}$. Recall the definition of $\hat{t}(\epsilon)$ in eq. (G.38).

First, using Lemma G.14, we know that for sufficiently small $\delta$, we have

$$\sup_{x \in [-2\epsilon, 2\epsilon]} \mathbb{P}\Big(A_1^\times(\epsilon, \delta, \eta)\Big) = o(\eta^N) \tag{G.72}$$

where

$$A_1^\times(\epsilon, \delta, \eta) = \Big\{ \exists n < T_1^\eta(\delta) \ s.t. \ |X_n^\eta(x)| > 3\epsilon \Big\}.$$

Define event

$$A_2^\times(\widetilde{\epsilon}, \delta, \eta) \triangleq \Big\{ \exists j = 2, \cdots, l^* \ s.t. \ \max_{k=1,2,\cdots,T_j^\eta(\delta) - T_{j-1}^\eta(\delta) - 1} \eta | Z_{\bar{T}_{j-1}^\eta(\delta)+1}^{\leq \delta, \eta} + \cdots + Z_{\bar{T}_{j-1}^\eta(\delta)+k}^{\leq \delta, \eta} | > \widetilde{\epsilon} \Big\}.$$

From Lemma G.12, we know that for sufficiently small $\delta > 0$,

$$\mathbb{P}\Big(A_2^\times(\widetilde{\epsilon}, \delta, \eta)\Big) = o(\eta^N). \tag{G.73}$$

From now on, we only consider such $\delta$ that eq. (G.72)eq. (G.73) hold.

On event $\left( A_1^\times \cup A_2^\times \right)^c \cap \{\sigma(\eta) < R(\epsilon, \eta)\} \cap \{\sigma(\eta) > T_1^\eta(\delta)\}$, we must have $\sigma(\eta) > T_1^\eta(\delta)$ and

$$T_2^\eta(\delta) \wedge \sigma(\eta) - T_1^\eta(\delta) < 2\hat{t}(\epsilon)/\eta.$$

Otherwise, due to Lemma G.6 and G.9, we know that at step $\widetilde{t} = T_1^\eta(\delta) + \lfloor \hat{t}(\epsilon)/\eta \rfloor$, we have

$$|X_{\widetilde{t}}^\eta| < 2\epsilon, \text{ and } |X_n^\eta| \le \bar{\epsilon} \ \forall n \le \widetilde{t}$$

for any sufficiently small $\eta$. By repeating this argument inductively, we obtain the following result: define

$$J = \min\{j = 1, 2, \cdots : \sigma(\eta) \in [T_j^\eta(\delta), T_{j+1}^\eta(\delta))\},$$

then on event $\left(A_1^\times \cup A_2^\times\right)^c \cap \{\sigma(\eta) < R(\epsilon, \eta), \ \sigma(\eta) < T_{l^*}^\eta(\delta)\}$, we must have

$$T_j^\eta(\delta) \wedge \sigma(\eta) - T_{j-1}^\eta(\delta) \wedge \sigma(\eta) < 2\hat{t}(\epsilon)/\eta \ \forall j = 2, 3, \cdots, J. \tag{G.74}$$

Furthermore, using this bound and Lemma G.9, we know that on event $\left(A_1^\times \cup A_2^\times\right)^c \cap \{\sigma(\eta) < R(\epsilon, \eta), \ \sigma(\eta) < T_{l^*}^\eta(\delta)\}$,

- $|X_{T_j^\eta(\delta)}^\eta| \le |X_{T_{j-1}^\eta(\delta)}^\eta| + b + \epsilon + \bar{\epsilon}$ for all $j = 2, 3, J - 1$,

- $|X_{\sigma(\eta)}^\eta| \le |X_{T_{J-1}^\eta(\delta)}^\eta| + \epsilon + \bar{\epsilon}$

However, this implies

$$|X_{\sigma(\eta)}^\eta| \le l^*(\bar{\epsilon} + \epsilon) + (l^* - 1)b < r$$

and contradicts the definition of $\sigma(\eta)$. In summary,

$$\sup_{x \in [-2\epsilon, 2\epsilon]} \mathbb{P}_x\left(\sigma(\eta) < R(\epsilon, \eta), \ \sigma(\eta) < T_{l^*}^\eta(\delta)\right) \le \mathbb{P}\left(A_1^\times(\epsilon, \delta, \eta) \cup A_2^\times(\widetilde{\epsilon}, \delta, \eta)\right) = o(\eta^N).$$

$\square$

The following two results follow immediately from the proof above, especially the inductive argument leading to bound eq. (G.74), and we state them without repeating the deatils of the proof.

**Corollary G.18.** *Given $\epsilon \in (0, \bar{\epsilon}), N > 0$, the following claim holds for any sufficiently small $\delta > 0$:*

$$\sup_{x \in [-2\epsilon, 2\epsilon]} \mathbb{P}_x\left(T_{l^*}^\eta(\delta) \le \sigma(\eta) \wedge R(\epsilon, \eta), \text{ and } \exists j = 2, 3, \cdots, l^* \text{ s.t. } T_j^\eta(\delta) - T_{j-1}^\eta(\delta) > 2\hat{t}(\epsilon)/\eta\right)$$

$$= o(\eta^N) \quad \text{as } \eta \downarrow 0.$$

**Corollary G.19.** *Given $\epsilon \in (0, \bar{\epsilon}), N > 0$, the following claim holds for any sufficiently small $\delta > 0$:*

$$\sup_{x \in [-2\epsilon, 2\epsilon]} \mathbb{P}_x\left(R(\epsilon, \eta) < T_{l^*}^\eta(\delta) \wedge \sigma(\eta), \ R(\epsilon, \eta) - T_1^\eta(\delta) > 2l^*\hat{t}(\epsilon)/\eta\right) = o(\eta^N)$$

*as $\eta \downarrow 0$.*

In the next result, we show that, if the inter-arrival time between some large jumps are too long, or some large jumps are still not *large enough*, then it is very unlikely that the SGD iterates could escape at the time of $l^*-$th large jump (or even get close enough to the boundary of the attraction field).

**Lemma G.20.** *Given $\epsilon \in (0, \bar{\epsilon})$ and any $N > 0$, the following holds for all $\delta > 0$ that are sufficiently small:*

$$\sup_{x \in [-2\epsilon, 2\epsilon]} \mathbb{P}_x(B_2^\times(\epsilon, \delta, \eta)) = o(\eta^N).$$

*where*

$$B_2^\times(\epsilon, \delta, \eta) = \{T_{l^*}^\eta(\delta) \le \sigma(\eta) \wedge R(\epsilon, \eta)\} \cap \Big\{\exists j = 2, 3, \cdots, l^* \text{ s.t. } T_j^\eta(\delta) - T_{j-1}^\eta(\delta) > \bar{t}/\eta$$

$$\text{or } \exists j = 1, 2, \cdots, l^* \text{ s.t. } \eta|W_j^\eta(\delta)| \le \bar{\delta}\Big\} \cap \{|X_{T_{l^*}^\eta}^\eta| \ge r - \bar{\epsilon}\}.$$

*Proof.* Let $A_1^\times$, $A_2^\times$ be the events defined in the proof of Lemma G.17. Based on the given $\epsilon > 0$, fix some $\widetilde{\epsilon} = \frac{\epsilon}{4\exp(2M\hat{t}(\epsilon))}$.

Let $J = \min\{j = 2, 3, \cdots : T_j^\eta(\delta) - T_{j-1}^\eta(\delta) > \bar{t}/\eta\}$. On event $\left(A_1^\times(\epsilon, \delta, \eta) \cup A_2^\times(\widetilde{\epsilon}, \delta, \eta)\right)^c \cap \{J \le l^*\}$, from Lemma G.6 and G.9 and the definition of constant $\bar{t}$, we know that

- $|X_{T_j^\eta(\delta)}^\eta| \le |X_{T_{j-1}^\eta(\delta)}^\eta| + b + \epsilon + \bar{\epsilon}$ for all $j = 2, 3, J - 1$;

- $|X_{T_J^\eta(\delta)}^\eta| \le 2\bar{\epsilon}$

- $|X_n^\eta| < s - \bar{\epsilon}$ $\forall n \le T_J^\eta(\delta)$

Now starting from step $T_J^\eta(\delta)$, by using Lemma G.6 and G.9 again one can see that

- $|X_{T_j^\eta(\delta)}^\eta| \le |X_{T_{j-1}^\eta(\delta)}^\eta| + b + \epsilon + \bar{\epsilon}$ for all $j = J + 1, \cdots, l^*$.

Combining these results, we have that $|X_{T_{l^*}^\eta}^\eta| < r - \bar{\epsilon}$ on event $\left(A_1^\times(\epsilon, \delta, \eta) \cup A_2^\times(\widetilde{\epsilon}, \delta, \eta)\right)^c \cap \{J \le l^*\}$.

Next, define $J' = \min\{j = 1, 2, \cdots; \eta|W_j^\eta(\delta)| \le \bar{\delta}\}$. Similarly, on event $\left(A_1^\times(\epsilon, \delta, \eta) \cup A_2^\times(\widetilde{\epsilon}, \delta, \eta)\right)^c \cap \{J > l^*\} \cap \{J' \le l^*\}$, using Lemma G.6 and G.9 again one can see that

- $|X_{T_j^\eta(\delta)}^\eta| \le |X_{T_{j-1}^\eta(\delta)}^\eta| + b + \epsilon + \bar{\epsilon}$ for all $j = 1, 2, \cdots, l^*, j \ne J'$;

- $|X_{T_J^\eta(\delta)}^\eta| \le |X_{T_{J-1}^\eta(\delta)}^\eta| + \bar{\delta} + \epsilon + \bar{\epsilon}$ for all $j = 1, 2, \cdots, l^*, j \ne J'$.

Since $\bar{\delta} \in (0, \bar{\epsilon})$, we have $|X_{T_{l^*}^\eta}^\eta| < r - \bar{\epsilon}$ on this event.

In summary, the following bound

$$\sup_{x \in [-2\epsilon, 2\epsilon]} \mathbb{P}_x(B_2^\times) \le \mathbb{P}(A_1^\times(\epsilon, \delta, \eta) \cup A_2^\times(\widetilde{\epsilon}, \delta, \eta)) = o(\eta^N)$$

holds for any $\delta$ that is sufficiently small, which is established in Lemma G.17. This conclude the proof. $\qquad\square$

In the next lemma, we show that, starting from the local minimum, it is unlikely that the SGD iterates will be right at the boundary of the attraction field after $l^*$ large jumps. Recall that there are $n_{\min}$ attraction fields on $f$, and excluding $s_0 = -\infty$, $s_{n_{\min}} = \infty$ the remaining points $s_1, \cdots, s_{n_{\min}-1}$ are the boundaries of the attraction fields.

**Lemma G.21.** *There exists a function $\Psi(\cdot) : \mathbb{R}^+ \mapsto \mathbb{R}^+$ satisfying $\lim_{\epsilon \downarrow 0} \Psi(\epsilon) = 0$ such that the following claim folds. Given $\epsilon \in \left(0, \bar{\epsilon}/(3\bar{\rho} + 3\widetilde{\rho} + 9)\right)$, we have*

$$\limsup_{\eta \downarrow 0} \frac{\sup_{x \in [-2\epsilon, 2\epsilon]} \mathbb{P}_x(B_3^\times(\epsilon, \delta, \eta))}{\left(H(1/\eta)/\eta\right)^{l^*-1}} \le \delta^\alpha \Psi(\epsilon)$$

*for all $\delta$ sufficiently small, where $\bar{\rho}$ and $\widetilde{\rho}$ are the constants defined in Corollary G.7 and G.10, and the event is defined as*

$$B_3^\times(\epsilon, \delta, \eta)$$
$$= \left\{T_{l^*}^\eta(\delta) \le \sigma(\eta) \wedge R(\epsilon, \eta)\right\} \cap \left\{\exists k \in [n_{min} - 1] \text{ such that } X_{T_{l^*}^\eta(\delta)}^\eta \in [s_k - \epsilon, s_k + \epsilon]\right\}.$$

*Proof.* Let $A_1^\times, A_2^\times$ be the events defined in the proof of Lemma G.17. Based on the given $\epsilon > 0$, fix some $\widetilde{\epsilon} = \frac{\epsilon}{4\exp(2M\hat{t}(\epsilon))}$. Fix some $N > \alpha l^*$.

Choose $\delta$ small enough so that claim in Lemma G.20 holds for the $\epsilon$ prescribed. Using the same arguments in Lemma G.20, we have the following inclusion of events:

$$B_3^\times(\epsilon, \delta, \eta) \cap \Big( A_1^\times(\epsilon, \delta, \eta) \cup A_2^\times(\widetilde{\epsilon}, \delta, \eta) \Big)^c$$

$$\subseteq \Big\{ \forall j = 2, 3, \cdots, l^*,\ T_j^\eta(\delta) - T_{j-1}^\eta(\delta) \le \bar{t}/\eta \Big\} \cap \Big\{ \forall j = 1, 2, 3, \cdots, l^*,\ \eta|W_1^\eta(\delta)| > \bar{\delta} \Big\}.$$

Therefore, on event $B_3^\times(\epsilon, \delta, \eta) \cap \Big( A_1^\times(\epsilon, \delta, \eta) \cup A_2^\times(\widetilde{\epsilon}, \delta, \eta) \Big)^c$, we can apply Corollary G.7 and G.10 and conclude that $Z_{T_1^\eta(\delta)}$ has $(-\bar{\epsilon}, \delta, \eta)-$overflow, and its $(-\bar{\epsilon}, \delta, \eta)-$overflow endpoint lies

$$(s_k - 3(\bar{\rho} + \widetilde{\rho} + 3)\epsilon, s_k + 3(\bar{\rho} + \widetilde{\rho} + 3)\epsilon)$$

for some $k \in [n_{\min} - 1]$. Using Lemma G.15 and Corollary G.16, we have that (for any sufficiently small $\eta$)

$$\mathbb{P}\Big( B_3^\times(\epsilon, \delta, \eta) \cap \Big( A_1^\times(\epsilon, \delta, \eta) \cup A_2^\times(\widetilde{\epsilon}, \delta, \eta) \Big)^c \Big)$$

$$\le \delta^\alpha \Big( \frac{H(1/\eta)}{\eta} \Big)^{l^*-1} \cdot \sum_{k=1}^{n_{\min}-1} \mu\Big( E(-\bar{\epsilon}) \cap h^{-1}\Big( (s_k - \hat{\epsilon}, s_k + \hat{\epsilon}) \Big) \Big)$$

where $\hat{\epsilon} = 3(\bar{\rho} + \widetilde{\rho} + 3)\epsilon$. Besides, as established in the proof of Lemma G.17, we have

$$\mathbb{P}\Big( A_1^\times(\epsilon, \delta, \eta) \cup A_2^\times(\widetilde{\epsilon}, \delta, \eta) \Big) = o(\eta^N)$$

for all sufficiently small $\delta$. In conclusion, we only need to choose

$$\Psi(\epsilon) = \sum_{k=1}^{n_{\min}-1} \mu\Big( E(-\bar{\epsilon}) \cap h^{-1}\big( (s_k - 3(\bar{\rho} + \widetilde{\rho} + 3)\epsilon, s_k + 3(\bar{\rho} + \widetilde{\rho} + 3)\epsilon) \big) \Big).$$

To conclude the proof, just note that by combining the continuity of measure with the conditional probability argument leading to eq. (G.69), we can show that $\lim_{\epsilon \downarrow 0} \Psi(\epsilon) = 0$. $\qquad\square$

Lastly, we establish the lower bound for the probability of the *most likely* way for SGD iterates to exit the current attraction field: making $l^*$ large jumps in a relatively short period of time. Recall that $\bar{\epsilon}$ is the fixed constant in eq. (G.26)-eq. (G.29).

**Lemma G.22.** *Given $\epsilon \in (0, \bar{\epsilon}/3)$, it holds for any sufficiently small $\delta > 0$ such that*

$$\liminf_{\eta \downarrow 0} \frac{\inf_{|x| \le 2\epsilon} \mathbb{P}_x(A^\circ(\epsilon, \delta, \eta))}{\big( H(1/\eta)/\eta \big)^{l^*-1}} \ge c_* \delta^\alpha$$

*where the event is defined as*

$$A^\circ(\epsilon, \delta, \eta) \triangleq \Big\{ \sigma(\eta) < R(\epsilon, \eta),\ \sigma(\eta) = T_{l^*}^\eta(\delta),\ X_{T_{l^*}^\eta}^\eta \notin [s_- - \epsilon, s_+ + \epsilon] \Big\}$$

$$\cap \Big\{ T_j^\eta(\delta) - T_{j-1}^\eta(\delta) \le \frac{\bar{\epsilon}}{2M}\lceil 1/\eta \rceil\ \forall j = 2, 3, \cdots, l^* \Big\}$$

*and the constant*

$$c_* = \frac{1}{2}\Big( \frac{1}{2b} \Big)^{l^*\alpha} \Big( \frac{\bar{\epsilon}}{4M} \Big)^{l^*-1}$$

*is strictly positive and does not vary with $\epsilon, \delta$.*

*Proof.* Let $A_1^\times, A_2^\times$ be the events defined in the proof of Lemma G.17. Fix some $N$ such that $N > \alpha l^*$. Based on the given $\epsilon > 0$, fix some $\widetilde{\epsilon} = \frac{\epsilon}{4\exp(2M\hat{t}(\epsilon))}$. We only consider $\delta < M$. Furthermore, choose $\delta$ small enough so that eq. (G.72) and eq. (G.73) hold for the chosen $N$ and $\epsilon$. Also, we only consider $\eta$ small enough so that $\eta M < b \wedge \bar{\epsilon}$.

Due to eq. (G.26)-eq. (G.29), we can, without loss of generality, assume that $r = s_+$, and in this case we will have

$$l^* b - 100 l^* \bar{\epsilon} > s_+ + 100 l^* \bar{\epsilon}.$$

Under this assumption, we will now focus on providing a lower bound for the following event that describes the exit from the right side of $\Omega$ (in other words, by crossing $s_+$)

$$A^\circ_\to(\epsilon, \delta, \eta) \triangleq \left\{ \sigma(\eta) < R(\epsilon, \eta),\ \sigma(\eta) = T^\eta_{l^*}(\delta),\ X^\eta_{T^\eta_{l^*}} > s_+ + \epsilon \right\}$$
$$\cap \left\{ T^\eta_j(\delta) - T^\eta_{j-1}(\delta) \leq \frac{\bar{\epsilon}}{2M} \lceil 1/\eta \rceil\ \forall j = 2, 3, \cdots, l^* \right\}.$$

First, define event

$$A^\circ_3(\delta, \eta) = \left\{ W^\eta_j(\delta) \geq 2b\ \forall j = 1, \cdots, l^*,\ T^\eta_j(\delta) - T^\eta_{j-1}(\delta) \leq \frac{\bar{\epsilon}}{2M} \lceil 1/\eta \rceil\ \forall j = 2, \cdots, l^* \right\},$$

and observe some facts on event $A^\circ_3(\delta, \eta) \cap \left( A^\times_1(\epsilon, \delta, \eta) \cup A^\times_2(\widetilde{\epsilon}, \delta, \eta) \right)^c$.

- $|X^\eta_k| \leq 3\epsilon \forall n < T^\eta_1(\delta)$; (due to $A^\times_1$ not occurring)

- $X^\eta_{T^\eta_1(\delta)} \in [b - 3\epsilon, b + 3\epsilon]$; (due to $W^\eta_1 \geq 2b$ and the effect of gradient clipping at step $T^\eta_1$, as well as the fact that $X^\eta_{T^\eta_1 - 1} \in [-3\epsilon, 3\epsilon]$ from the previous bullet point)

- Due to $|f'(\cdot)| \leq M$ and $\delta < M$, one can see that (for any $n \geq 1$)
$$\sup_{x \in [-L, L]} |\eta f'(x)| + |\eta Z^{\leq \delta, \eta}_n| \leq 2\eta M;$$
this provides an upper bound for the change in SGD iterates at each step, and gives us
$$X^\eta_n \in [b - 3\epsilon - \bar{\epsilon}, b + 3\epsilon + \bar{\epsilon}]\ \forall T^\eta_1(\delta) \leq n < T^\eta_2(\delta)$$
where we also used $T^\eta_2(\delta) - T^\eta_1(\delta) \leq \frac{\bar{\epsilon}}{2M} \lceil 1/\eta \rceil$

- Therefore, at the arrival time of the second large jump, we must have $X^\eta_{T^\eta_2(\delta)} \geq 2b - 3\epsilon - \bar{\epsilon}$;

- By repeating the argument above inductively, we can show that (for all $j = 1, 2, \cdots, l^*$)
$$X^\eta_n \in [(j-1)b - 3\epsilon - (j-1)\bar{\epsilon}, (j-1)b + 3\epsilon + (j-1)\bar{\epsilon}]\ \forall T^\eta_{j-1} \leq n < T^\eta_j$$
$$X^\eta_{T^\eta_j} \in [jb - 3\epsilon - (j-1)\bar{\epsilon}, jb + 3\epsilon + (j-1)\bar{\epsilon}];$$
In particular, we know that $X^\eta_n \in \Omega$ for any $n < T^\eta_{l^*}$ (so the exit does not occur before $T^\eta_{l^*}$), and at the arrival of the $l^*$−th large jump, we have (using $3\epsilon < \bar{\epsilon}$)
$$X^\eta_{T^\eta_{l^*}(\delta)} \geq l^* b - l^* \bar{\epsilon} > s_+ + \epsilon.$$

In summary, we have shown that

$$A^\circ_3(\delta, \eta) \cap \left( A^\times_1(\epsilon, \delta, \eta) \cup A^\times_2(\widetilde{\epsilon}, \delta, \eta) \right)^c \subseteq A^\circ_\to(\epsilon, \delta, \eta).$$

To conclude the proof, just notice that (for sufficiently small $\eta$)

$$\mathbb{P}\left( A^\circ_3(\delta, \eta) \cap \left( A^\times_1(\epsilon, \delta, \eta) \cup A^\times_2(\widetilde{\epsilon}, \delta, \eta) \right)^c \right)$$
$$\geq \mathbb{P}(A^\circ_3(\delta, \eta)) - \mathbb{P}(A^\times_1(\epsilon, \delta, \eta)) - \mathbb{P}(A^\times_2(\widetilde{\epsilon}, \delta, \eta))$$
$$\geq \mathbb{P}(A^\circ_3(\delta, \eta)) - \eta^N \quad \text{due to eq. (G.72) and eq. (G.73)}$$
$$\geq \left( \frac{H(2b/\eta)}{H(\delta/\eta)} \right)^{l^*} \left( \frac{\bar{\epsilon}}{4M} H(\delta/\eta)/\eta \right)^{l^* - 1} - \eta^N \quad \text{due to Lemma G.4}$$
$$\geq 2c_* \delta^\alpha (H(1/\eta)/\eta)^{l^* - 1} - \eta^N \quad \text{for all } \eta \text{ sufficiently small, due to } H \in \mathcal{RV}_{-\alpha}$$
$$\geq c_* \delta^\alpha (H(1/\eta)/\eta)^{l^* - 1}.$$

$\square$

In order to present the main result of this section, we need to take into account the loss landscape outside of the current attraction field $\Omega$. Recall that there are $n_{\min}$ attraction fields on $f$. For all the attraction fields different from $\Omega$, we call them $(\widetilde{\Omega}_k)_{k=1}^{n_{\min}-1}$ where, for each $k \in [n_{\min} - 1]$, the attraction field $\widetilde{\Omega}_k = (s_k^-, s_k^+)$ with the corresponding local minimum located at $\widetilde{m}_k$. Also, recall that $\sigma(\eta)$ is the first time $X_n^\eta$ exits from $\Omega$. Building upon these concepts, we can define a stopping time

$$\tau(\eta, \epsilon) \triangleq \min\{n \geq 0 : X_n^\eta \in \bigcup_{k=1}^{n_{\min}-1} [\widetilde{m}_k - 2\epsilon, \widetilde{m}_k + 2\epsilon]\} \tag{G.75}$$

as the first time the SGD iterates visit a minimizer in an attraction field that is different from $\Omega$. Besides, let index $J_\sigma(\eta)$ be such that

$$J_\sigma(\eta) = j \iff X_{\sigma(\eta)}^\eta \in \widetilde{\Omega}_j \quad \forall j \in [n_{\min} - 1]. \tag{G.76}$$

In other words, it is the label of the attraction field that $X_n^\eta$ escapes to. Lastly, define

$$\lambda(\eta) \triangleq H(1/\eta)\Big(H(1/\eta)/\eta\Big)^{l^*-1}, \tag{G.77}$$

$$\nu^\Omega \triangleq \mu\big(E(0)\big), \tag{G.78}$$

$$\nu_k^\Omega \triangleq \mu\big(E(0) \cap h^{-1}(\widetilde{\Omega}_k)\big) \quad \forall k \in [n_{\min} - 1]. \tag{G.79}$$

For definitions of the measure $\mu$, set $E$, and mapping $h$, see eq. (G.64) and eq. (G.65).

Now we are ready to state Proposition G.23, the most important technical tool in this section. In eq. (G.80) and eq. (G.81), we provide upper and lower bounds for the joint distribution of first exit time $\sigma$ and the label $J_\sigma$ indexing the attraction field we escape to; it is worth noticing that the claims hold uniformly for all $u > C$. In eq. (G.82) and eq. (G.83), we provide upper and lower bounds for the joint distribution of when we first visit a different local minimum (which is equal to $\tau$) and which one we visit (indicated by $X_\tau^\eta$). The similarity between eq. (G.80) eq. (G.81) and eq. (G.82)eq. (G.83) suggests a strong correlation between the behavior of the SGD iterates at time $\sigma(\eta)$ and that of time $\tau(\eta, \epsilon)$, and this is corroborated by eq. (G.84): we show that it is almost always the case that $\tau$ is very close to $\sigma$, and on the short time interval $[\sigma(\eta), \tau(\eta, \epsilon)]$ the SGD iterates stay within the same attraction field.

**Proposition G.23.** *Given $C > 0$ and some $k' \in [n_{min} - 1]$, the following claims hold for all $\epsilon > 0$ that is sufficiently small:*

$$\limsup_{\eta\downarrow 0} \sup_{u\in(C,\infty)} \sup_{x\in[-2\epsilon,2\epsilon]} \mathbb{P}_x\Big(\nu^\Omega\lambda(\eta)\sigma(\eta) > u,\ J_\sigma(\eta) = k'\Big)$$
$$\leq 2C + \exp(-(1-C)^3 u)\frac{\nu_{k'}^\Omega + C}{\nu^\Omega}, \tag{G.80}$$

$$\liminf_{\eta\downarrow 0} \inf_{u\in(C,\infty)} \inf_{x\in[-2\epsilon,2\epsilon]} \mathbb{P}_x\Big(\nu^\Omega\lambda(\eta)\sigma(\eta) > u,\ J_\sigma(\eta) = k'\Big)$$
$$\geq -2C + \exp(-(1+C)^3 u)\frac{\nu_{k'}^\Omega - C}{\nu^\Omega}, \tag{G.81}$$

$$\limsup_{\eta\downarrow 0} \sup_{u\in(C,\infty)} \sup_{x\in[-2\epsilon,2\epsilon]} \mathbb{P}_x\Big(\nu^\Omega\lambda(\eta)\tau(\eta,\epsilon) > u,\ X_{\tau(\eta,\epsilon)}^\eta \in B(\widetilde{m}_{k'}, 2\epsilon)\Big)$$
$$\leq 4C + \exp(-(1-C)^3 u)\frac{\nu_{k'}^\Omega + C}{\nu^\Omega}, \tag{G.82}$$

$$\liminf_{\eta\downarrow 0} \inf_{u\in(C,\infty)} \inf_{x\in[-2\epsilon,2\epsilon]} \mathbb{P}_x\Big(\nu^\Omega\lambda(\eta)\tau(\eta,\epsilon) > u,\ X_{\tau(\eta,\epsilon)}^\eta \in B(\widetilde{m}_{k'}, 2\epsilon)\Big)$$
$$\geq -4C + \exp(-(1+C)^3 u)\frac{\nu_{k'}^\Omega - C}{\nu^\Omega}, \tag{G.83}$$

$$\liminf_{\eta\downarrow 0} \inf_{x\in[-2\epsilon,2\epsilon]} \mathbb{P}_x\Big(\lambda(\eta)\big(\tau(\eta,\epsilon) - \sigma(\eta)\big) < C,\ X_n^\eta \in \widetilde{\Omega}_{J_\sigma(\eta)}\ \forall n \in [\sigma(\eta), \tau(\eta,\epsilon)]\Big)$$
$$\geq 1 - C. \tag{G.84}$$

Before presenting the proof to Proposition G.23, we make some preparations. First, we introduce stopping times (for all $k \geq 1$)

$$\tau_k(\epsilon, \delta, \eta) = \min\{n > \widetilde{\tau}_{k-1}(\epsilon, \delta, \eta) : \eta|Z_n| > \delta\}$$
$$\widetilde{\tau}_k(\epsilon, \delta, \eta) = \min\{n \geq \tau_k(\epsilon, \delta, \eta) : |X_n^\eta| \leq 2\epsilon\}$$

with the convention that $\tau_0(\epsilon, \delta, \eta) = \widetilde{\tau}_0(\epsilon, \delta, \eta) = 0$. The intuitive interpretation is as follows. For the fixed $\epsilon$ we treat $[-2\epsilon, 2\epsilon]$ as a small neighborhood of the local minimum of the attraction field $\Omega$. All the $\widetilde{\tau}_k$ partitioned the entire timeline into different *attempts* of escaping $\Omega$. The interval $[\widetilde{\tau}_{k-1}, \widetilde{\tau}_k]$ can be viewed as the $k-$th *attempt*. If for $\sigma(\eta)$, the first exit time defined in eq. (G.70), we have $\sigma(\eta) > \widetilde{\tau}_k$, then we consider the $k-$th attempt of escape as a *failure* because the SGD iterates returned to this small neighborhood of the local minimum again without exiting the attraction field. On the other hand, the stopping times $\tau_{k-1}$ indicate the arrival time of the first large jump during the $k-$th *attempt*. The proviso that $\widetilde{\tau}_k \geq \tau_{k-1}$ can be interpreted, intuitively, as that an *attempt* is considered failed only if, after some significant efforts to exit (for instance, a large jump) has been observed, the SGD iterates still returned to the small neighborhood $[-2\epsilon, 2\epsilon]$. Regarding the notations, we add a remark that when there is no ambiguity we will drop the dependency on $\epsilon, \delta, \eta$ and simply write $\tau_k, \widetilde{\tau}_k$.

To facilitate the characterization of events during each *attempt*, we introduce the following definitions. First, for all $k \geq 1$, let

$$\mathbf{j}_k \triangleq \#\{n = \tau_{k-1}(\epsilon, \delta, \eta), \tau_{k-1}(\epsilon, \delta, \eta) + 1, \cdots, \widetilde{\tau}_k(\epsilon, \delta, \eta) \wedge \sigma(\eta) : \eta|Z_n| > \delta\}$$

be the number of large jumps during the $k-$th *attempt*. Two implications of this definition:

- First, for any $k$ with $\sigma(\eta) < \widetilde{\tau}_k$, we have $\mathbf{j}_k = 0$. Note that this proposition concerns the dynamics of SGD up until $\sigma(\eta)$, the first time the SGD iterates escaped from $\Omega$, so there is no need to consider an attempt that is after $\sigma(\eta)$, and we will not do so in the analysis below;

- Besides, the random variable $\mathbf{j}_k$ is measurable w.r.t. $\mathcal{F}_{\widetilde{\tau}_k \wedge \sigma(\eta)}$, the stopped $\sigma-$algebra generated by the stopping time $\widetilde{\tau}_k \wedge \sigma(\eta)$.

Furthermore, for each $k = 1, 2, \cdots$, let

$$T_{k,1}(\epsilon, \delta, \eta) = \tau_{k-1}(\epsilon, \delta, \eta) \wedge \sigma(\eta),$$
$$T_{k,j}(\epsilon, \delta, \eta) = \min\{n > T_{k,j-1}(\epsilon, \delta, \eta) : \eta|Z_n| > \delta\} \wedge \sigma(\eta) \wedge \widetilde{\tau}_k \ \ \forall j \geq 2,$$
$$W_{k,j}(\epsilon, \delta, \eta) = Z_{T_{k,j}(\epsilon, \delta, \eta)} \ \ \forall j \geq 1$$

with the convention $T_{k,0}(\epsilon, \delta, \eta) = \widetilde{\tau}_{k-1}(\epsilon, \delta, \eta)$. Note that for any $k \geq 1, j \geq 1, T_{k,j}$ is a stopping time. Besides, from the definition of $\mathbf{j}_k$, one can see that

$$\widetilde{\tau}_{k-1} + 1 \leq T_{k,j} \leq \widetilde{\tau}_k \wedge \sigma(\eta) \ \ \forall j \in [\mathbf{j}_k], \tag{G.85}$$

and the sequences $\left(T_{k,j}\right)_{j=1}^{\mathbf{j}_k}$ and $\left(W_{k,j}\right)_{j=1}^{\mathbf{j}_k}$ are the arrival times and sizes of *large* jumps during the $k-$th attempt, respectively. Again, when there is no ambiguity we will drop the dependency on $\epsilon, \delta, \eta$ and simply write $T_{k,j}$ and $W_{k,j}$.

In order to prove Proposition G.23, we analyze the most likely scenario that the exit from $\Omega$ would happen. Specifically, we will introduce a series of events with superscript $\times$ or $\circ$, where $\times$ indicates that the event is *atypical* or unlikely to happen and $\circ$ means that it is a *typical* event and is likely to be observed before the first exit from the attraction field $\Omega$. Besides, the subscript $k$ indicates that the event in discussion concerns the dynamics of SGD during the $k-$th attempt. Our goal is to show that for some event $\mathbf{A}^\times(\epsilon, \delta, \eta)$ its probability becomes sufficiently small as learning rate $\eta$ tends to 0, so the escape from $\Omega$ almost always occurs in the manner described by $(\mathbf{A}^\times(\epsilon, \delta, \eta))^c$. In particular, the definition of this *atypical* scenario $\mathbf{A}^\times$ involves the union of some *atypical* events $\mathbf{A}_k^\times, \mathbf{B}_k^\times$ that occur in the $k-$th attempt. In other words, the intuition of $\mathbf{A}^\times$ is that something *abnormal* happened during one of the attempts before the final exit.

Here is one more comment for the general naming convention of these events. Events with label $\mathbf{A}$ often describe the "efforts" made in an attempt to get out of $\Omega$ (such as large noises), while those with label $\mathbf{B}$ concern how the SGD iterates return to $[-2\epsilon, 2\epsilon]$ (and how this attempt fails). For instance,

$\mathbf{A}_k^\times$ discusses the unlikely scenario before $T_{k,l^*}$, the arrival of the $l^*-$th large jump in this attempt, while $\mathbf{B}_k^\times$ in general discusses the abnormal cases after $T_{k,l^*}$ and before the return to $[-2\epsilon, 2\epsilon]$. On the other hand, $\mathbf{A}_k^\circ$ describes a successful escape during $k-$th attempt, while $\mathbf{B}_k^\circ$ means that during this attempt the iterates return to without spending too much time.

Now we proceed and provide a formal definition and analysis of the aforementioned series of events. As building blocks, we inspect the process $(X_n^\eta)_{n\geq 1}$ at a even finer granularity, and bound the probability of some events $(\mathbf{A}_{k,i}^\times)_{i\geq 0}$, $(\mathbf{B}_{k,i}^\times)_{i\geq 1}$ detailing several cases that are *unlikely* to occur during the escape from or return to local minimum in the $k-$th attempt. First, for each $k \geq 1$, define the event

$$\mathbf{A}_{k,0}^\times(\epsilon, \delta, \eta) \triangleq \Big\{ \exists i = 0, 1, \cdots, l^* \wedge \mathbf{j}_k \ s.t.$$

$$\max_{j=T_{k,i}+1,\cdots,(T_{k,i+1}-1)\wedge\widetilde{\tau}_k\wedge\sigma(\eta)} \eta |Z_{T_{k,i}+1}^{\leq\delta,\eta} + \cdots + Z_j^{\leq\delta,\eta}| > \frac{\epsilon}{3\bar{\rho} + 3\widetilde{\rho} + 3} \Big\}. \qquad (G.86)$$

Intuitively speaking, the event characterizes the atypical scenario where, during the $k-$th attempt, there is some large fluctuations (compared to $\widetilde{\epsilon}$) between any of the first $l^*$ large jumps (or the first $\mathbf{j}_k$ large jumps in case that $\mathbf{j}_k < l^*$). Similarly, consider event (for all $k \geq 1$)

$$\mathbf{A}_{k,1}^\times(\epsilon, \delta, \eta) \triangleq \Big\{ \sigma(\eta) < \widetilde{\tau}_k, \ \mathbf{j}_k < l^* \Big\} \qquad (G.87)$$

that describes the atypical case where the exit occurs during the $k-$th attempt with less than $l^*$ large jumps. Next, for all $k \geq 1$ we have another atypical event (note that from eq. (G.85) we can see that, for any $j \geq 1, \mathbf{j}_k \geq j$ implies $T_{k,j} \leq \sigma(\eta) \wedge \widetilde{\tau}_k$)

$$\mathbf{A}_{k,2}^\times \triangleq \Big\{ \mathbf{j}_k \geq l^*, \ \exists j = 2, 3, \cdots, l^* \ s.t. \ T_{k,j} - T_{k,j-1} > 2\hat{t}(\epsilon)/\eta \Big\}. \qquad (G.88)$$

representing the case where we have at least $l^*$ large noises during the $k-$th attempt, but for some of the large noise (from the 2nd to the $l^*$-th), the inter-arrival time is unusually long. Moving on, we consider the following events (defined for all $k \geq 1$)

$$\mathbf{A}_{k,3}^\times \triangleq \Big\{ \mathbf{j}_k < l^*, \ \widetilde{\tau}_k < \sigma(\eta), \ \widetilde{\tau}_k - T_{k,1} > 2l^*\hat{t}(\epsilon)/\eta \Big\} \qquad (G.89)$$

that describes the atypical case where the $k-$th attempt failed but the return to the small neighborhood $[-2\epsilon, 2\epsilon]$ took unusually long time.

The following event also concerns the scenario where there are at least $l^*$ large noises during the $k-$th attempt:

$$\mathbf{A}_{k,4}^\times \triangleq \Big\{ \mathbf{j}_k \geq l^*, \ |X_{T_{k,l^*}}^\eta| \geq r - \bar{\epsilon}, \ \exists j = 1, 2, \cdots, l^* \ s.t. \ \eta|W_{k,j}| \leq \bar{\delta} \Big\}; \qquad (G.90)$$

specifically, it describes the atypical case where, during this attempt, right after the $l^*-$ large noise the SGD iterate is far enough from the local minimum yet some of the large noises are not that *large*. Lastly, by defining events

$$\mathbf{A}_{k,5}^\times \triangleq \Big\{ \mathbf{j}_k \geq l^*, \ T_{k,l^*} \leq \sigma(\eta) \wedge \widetilde{\tau}_k, \ X_{T_{k,l^*}}^\eta \in \bigcup_{j\in[n_{\min}-1]} [s_j - \epsilon, s_j + \epsilon] \Big\}, \qquad (G.91)$$

we analyze an atypical case where the SGD iterates arrive at somewhere too close to the boundaries of $\Omega$ at the arrival time of the $l^*$ large noise during this attempt. As an amalgamation of these atypical scenarios, we let

$$\mathbf{A}_k^\times(\epsilon, \delta, \eta) \triangleq \bigcup_{i=0}^5 \mathbf{A}_{k,i}^\times(\epsilon, \delta, \eta). \qquad (G.92)$$

Also, we analyze the probability of some events $(\mathbf{B}_k^\times)_{k\geq 1}$ that concern the SGD dynamics after the $l^*-$th large noise during the $k-$th attempt. Let us define

$$\mathbf{B}_{k,1}^\times(\epsilon, \delta, \eta) \triangleq \Big\{ \mathbf{j}_k \geq l^*, \ X_{T_{k,l^*}}^\eta \in [s_- + \epsilon, s_+ - \epsilon], \ T_{k,j} - T_{k,j-1} \leq 2\frac{\hat{t}(\epsilon)}{\eta} \ \forall j = 2, 3, \cdots, l^* \Big\}$$

$$\mathbf{B}_{k,2}^\times(\epsilon, \delta, \eta) \triangleq \{ \widetilde{\tau}_k - T_{k,l^*} > \rho(\epsilon)/\eta \} \cup \{ \sigma(\eta) < \widetilde{\tau}_k \}$$

$$\mathbf{B}_k^\times(\epsilon, \delta, \eta) \triangleq \mathbf{B}_{k,1}^\times \cap \mathbf{B}_{k,2}^\times \qquad (G.93)$$

where $\rho(\cdot)$ is the function in Lemma G.13. From the definition of $\mathbf{B}_k^\times$, in particular the inclusion of $\mathbf{B}_{k,2}^\times$, one can see that the intuitive interpretation of event $\mathbf{B}_k^\times$ is that the SGD iterates *did not return* to local minimum efficiently (or simply escaped from the attraction field) after the $l^*-$th large noise during the $k-$th attempt. In comparison, the following events will characterize what would typically happen during each attempt:

$$\mathbf{A}_k^\circ(\epsilon, \delta, \eta) \triangleq \{\mathbf{j}_k \geq l^*, \ \sigma(\eta) = T_{k,l^*}, \ X_{T_{k,l^*}}^\eta \notin [s_- - \epsilon, s_+ + \epsilon],$$

$$T_{k,j} - T_{k,j-1} \leq \frac{2\hat{t}(\epsilon)}{\eta} \ \forall j = 2, 3, \cdots, l^*\}, \tag{G.94}$$

$$\mathbf{B}_k^\circ(\epsilon, \delta, \eta) \triangleq \{\sigma(\eta) > \widetilde{\tau}_k, \ \widetilde{\tau}_k - T_{k,1} \leq \frac{2l^*\hat{t}(\epsilon) + \rho(\epsilon)}{\eta}\}. \tag{G.95}$$

Intuitively speaking, $\mathbf{A}_k^\circ$ tells us that the exit happened right at $T_{k,l^*}$, the arrival time of the $l^*-$th large noise during the $k-$th attempt, and $\mathbf{B}_k^\circ$ tells us that the first exit from $\Omega$ did not occur during the $k-$th attempt, and the SGD iterates returned to local minimum rather efficiently. All the preparations above allow use to define

$$\mathbf{A}^\times(\epsilon, \delta, \eta) \triangleq \bigcup_{k \geq 1} \left( \bigcap_{i=1}^{k-1} \left( \mathbf{A}_i^\times \cup \mathbf{B}_i^\times \cup \mathbf{A}_i^\circ \right)^c \right) \cap \left( \mathbf{A}_k^\times \cup \mathbf{B}_k^\times \right). \tag{G.96}$$

We need the next lemma in the proof of Proposition G.23. As mentioned earlier, the takeaway is that $\mathbf{A}^\times$ is indeed *atypical* in the sense that we will almost always observe $(\mathbf{A}^\times)^c$.

**Lemma G.24.** *Given any $C > 0$, the following claim holds for all $\epsilon > 0, \delta > 0$ sufficiently small:*

$$\limsup_{\eta \downarrow 0} \ \sup_{|x| \leq 2\epsilon} \ \mathbb{P}_x(\mathbf{A}^\times(\epsilon, \delta, \eta)) < C.$$

*Proof.* We fix some parameters for the proof. First, with out loss of generality we only consider $C \in (0, 1)$, and we fix some $N > \alpha l^*$. Next we discuss the valid range of $\epsilon$ for the claim to hold. We only consider $\epsilon > 0$ such that

$$\epsilon < \frac{\bar{\epsilon}}{6(\bar{\rho} + \widetilde{\rho} + 3)} \wedge \frac{\epsilon_0}{3}$$

where $\bar{\rho}$ and $\widetilde{\rho}$ are the constants in Corollary G.7 and Corollary G.10 respectively, and $\epsilon_0$ is the constant in eq. (G.8). Moreover, recall function $\Psi$ in Lemma G.21 and the constant $c_* > 0$ in Lemma G.22. Due to $\lim_{\epsilon \downarrow 0} \Psi(\epsilon) = 0$, it holds for all $\epsilon$ small enough such that

$$\frac{3\Psi(\epsilon)}{c_*} < C \tag{G.97}$$

In our proof we only consider $\epsilon$ small enough so the inequality above holds, and the claim in Lemma G.13 holds. Now we specify the valid range of parameter $\delta$ that will be used below:

- For all sufficiently small $\delta > 0$, the claim in Lemma G.14 will hold for the prescribed $\epsilon$ and with $N_0 = N$;

- For all sufficiently small $\delta > 0$, the claims in Lemma G.17, Corollary G.18, Corollary G.19 and Lemma G.20 will hold with the prescribed $\epsilon$ and $N$;

- For all sufficiently small $\delta > 0$, the inequalities in Lemma G.21 and G.22 will hold for the $\epsilon$ we fixed at the beginning.

We show that the claim holds for any $\epsilon, \delta$ small enough to satisfy the conditions above.

First, recall that

$$\mathbf{A}_{k,0}^\times(\epsilon, \delta, \eta) \triangleq \Big\{ \exists i = 0, 1, \cdots, l^* \wedge \mathbf{j}_k \ \text{s.t.}$$

$$\max_{j = T_{k,i}+1, \cdots, (T_{k,i+1}-1) \wedge \widetilde{\tau}_k \wedge \sigma(\eta)} \eta |Z_{\widetilde{T}_{k,i}+1}^{\leq \delta, \eta} + \cdots + Z_{\widetilde{j}}^{\leq \delta, \eta}| > \frac{\epsilon}{3\bar{\rho} + 3\widetilde{\rho} + 3} \Big\}.$$

Due to our choice of $\delta$ stated earlier and Lemma G.12, there exists some $\eta_0 > 0$ such that for all $\eta \in (0, \eta_0)$,

$$\mathbb{P}\big(\mathbf{A}_{k,0}^{\times}(\epsilon, \delta, \eta)\big) \leq \eta^N \ \ \forall k \geq 1. \tag{G.98}$$

Similarly, recall that $\mathbf{A}_{k,1}^{\times}(\epsilon, \delta, \eta) \triangleq \big\{\sigma(\eta) < \widetilde{\tau}_k, \ \mathbf{j}_k < l^*\big\}$. Let us temporarily focus on the first attempt (namely the case $k = 1$). From Lemma G.17 and our choice of $\epsilon$ and $\delta$, we know the existence of some $\eta_1 > 0$ such that

$$\sup_{|x| \leq 2\epsilon} \mathbb{P}_x\big(\mathbf{A}_{1,1}^{\times}(\epsilon, \delta, \eta)\big) \leq \eta^N \ \ \forall \eta \in (0, \eta_1). \tag{G.99}$$

Next, for $\mathbf{A}_{k,2}^{\times} \triangleq \big\{\mathbf{j}_k \geq l^*, \ \exists j = 2, 3, \cdots, l^* \ s.t. \ T_{k,j} - T_{k,j-1} > 2\hat{t}(\epsilon)/\eta\big\}$, from Corollary G.18 and our choice of $\delta$ at the beginning, we have the existence of some $\eta_2 > 0$ such that

$$\sup_{|x| \leq 2\epsilon} \mathbb{P}_x\big(\mathbf{A}_{1,2}^{\times}(\epsilon, \delta, \eta)\big) \leq \eta^N \ \ \forall \eta \in (0, \eta_2). \tag{G.100}$$

Moving on, for $\mathbf{A}_{k,3}^{\times} \triangleq \big\{\mathbf{j}_k < l^*, \ \widetilde{\tau}_k < \sigma(\eta), \ \widetilde{\tau}_k - T_{k,1} > 2l^*\hat{t}(\epsilon)/\eta\big\}$, due to Corollary G.19 and our choice of $\epsilon, \delta$, we have the existence of some $\eta_3 > 0$ such that

$$\sup_{|x| \leq 2\epsilon} \mathbb{P}_x\big(\mathbf{A}_{1,3}^{\times}(\epsilon, \delta, \eta)\big) \leq \eta^N \ \ \forall \eta \in (0, \eta_3). \tag{G.101}$$

As for $\mathbf{A}_{k,4}^{\times} \triangleq \big\{\mathbf{j}_k \geq l^*, \ |X_{T_{k,l^*}}^{\eta}| \geq r - \bar{\epsilon}, \ \exists j = 1, 2, \cdots, l^* \ s.t. \ \eta|W_{k,j}| \leq \bar{\delta}\big\}$, from Lemma G.20, one can see the existence of $\eta_4 > 0$ such that

$$\sup_{|x| \leq 2\epsilon} \mathbb{P}_x\big(\mathbf{A}_{1,4}^{\times}(\epsilon, \delta, \eta)\big) \leq \eta^N \ \ \forall \eta \in (0, \eta_4). \tag{G.102}$$

Lastly, for $\mathbf{A}_{k,5}^{\times} \triangleq \big\{\mathbf{j}_k \geq l^*, \ T_{k,l^*} \leq \sigma(\eta) \wedge \widetilde{\tau}_k, \ X_{T_{k,l^*}}^{\eta} \in \bigcup_{j \in [n_{\min}-1]}[s_j - \epsilon, s_j + \epsilon]\big\}$, from Lemma G.21 we see the existence of $\eta_5 > 0$ such that

$$\sup_{|x| \leq 2\epsilon} \mathbb{P}_x\big(\mathbf{A}_{1,5}^{\times}(\epsilon, \delta, \eta)\big) \leq 2\delta^{\alpha} \Psi(\epsilon)\Big(H(1/\eta)/\eta\Big)^{l^*-1} \ \ \forall \eta \in (0, \eta_5). \tag{G.103}$$

Recall that $\mathbf{A}_k^{\times}(\epsilon, \delta, \eta) = \bigcup_{i=0}^5 \mathbf{A}_{k,i}^{\times}(\epsilon, \delta, \eta)$. Also, for definitions of $\mathbf{B}_k^{\times}, \mathbf{A}_k^{\circ}, \mathbf{B}_k^{\circ}$, see eq. (G.93),eq. (G.94),eq. (G.95) respectively. Our next goal is to establish bounds regarding the probabilities of these events. First, if we consider the event $\bigcap_{j=1}^k (\mathbf{A}_j^{\times} \cup \mathbf{B}_j^{\times})^c \cap \mathbf{B}_j^{\circ}$, then the inclusion of the $(\mathbf{B}_j^{\circ})_{j=1}^k$ implies that during the first $k$ attempts the SGD iterates have never left the attraction field, so

$$\bigcap_{j=1}^k (\mathbf{A}_j^{\times} \cup \mathbf{B}_j^{\times})^c \cap \mathbf{B}_j^{\circ} = \big(\bigcap_{j=1}^k (\mathbf{A}_j^{\times} \cup \mathbf{B}_j^{\times})^c \cap \mathbf{B}_j^{\circ}\big) \cap \{\sigma(\eta) > \widetilde{\tau}_k\}.$$

Next, note that

$$\mathbb{P}_x\big(\mathbf{B}_k^{\times} \mid \bigcap_{j=1}^{k-1} (\mathbf{A}_j^{\times} \cup \mathbf{B}_j^{\times})^c \cap \mathbf{B}_j^{\circ}\big)$$

$$= \mathbb{P}_x\big(\mathbf{B}_{k,1}^{\times} \mid \bigcap_{j=1}^{k-1} (\mathbf{A}_j^{\times} \cup \mathbf{B}_j^{\times})^c \cap \mathbf{B}_j^{\circ}\big) \mathbb{P}_x\Big(\mathbf{B}_{k,2}^{\times} \mid \big(\bigcap_{j=1}^{k-1} (\mathbf{A}_j^{\times} \cup \mathbf{B}_j^{\times})^c \cap \mathbf{B}_j^{\circ}\big) \cap \mathbf{B}_{k,1}^{\times}\Big)$$

$$\leq \mathbb{P}_x\big(\mathbf{j}_k \geq l^*, \ T_{k,j} - T_{k,j-1} \leq \frac{2\hat{t}(\epsilon)}{\eta} \ \forall j = 2, 3, \cdots, l^* \mid \bigcap_{j=1}^{k-1} (\mathbf{A}_j^{\times} \cup \mathbf{B}_j^{\times})^c \cap \mathbf{B}_j^{\circ}\big)$$

$$\cdot \mathbb{P}_x\Big(\mathbf{B}_{k,2}^{\times} \mid \big(\bigcap_{j=1}^{k-1} (\mathbf{A}_j^{\times} \cup \mathbf{B}_j^{\times})^c \cap \mathbf{B}_j^{\circ}\big) \cap \mathbf{B}_{k,1}^{\times}\Big).$$

From the definition of the events $\mathbf{A}_j^\times, \mathbf{B}_j^\times, \mathbf{B}_j^\circ$, one can see that $\bigcap_{j=1}^{k-1}(\mathbf{A}_j^\times \cup \mathbf{B}_j^\times)^c \cap \mathbf{B}_j^\circ \in \mathcal{F}_{\widetilde{\tau}_{k-1} \wedge \sigma(\eta)}$, and on this event $\bigcap_{j=1}^{k-1}(\mathbf{A}_j^\times \cup \mathbf{B}_j^\times)^c \cap \mathbf{B}_j^\circ$ we have $\sigma(\eta) > \widetilde{\tau}_{k-1}$. So by applying strong Markov property at stopping time $\widetilde{\tau}_{k-1} \wedge \sigma(\eta)$, we have

$$
\mathbb{P}_x\Big(\mathbf{B}_k^\times \mid \bigcap_{j=1}^{k-1}(\mathbf{A}_j^\times \cup \mathbf{B}_j^\times)^c \cap \mathbf{B}_j^\circ\Big)
$$

$$
\leq \mathbb{P}\Big(T_j^\eta(\delta) - T_{j-1}^\eta(\delta) \leq 2\hat{t}(\epsilon)/\eta \ \forall j \in [l^*-1]\Big) \cdot \mathbb{P}_x\Big(\mathbf{B}_{k,2}^\times \ \Big| \ \big(\bigcap_{j=1}^{k-1}(\mathbf{A}_j^\times \cup \mathbf{B}_j^\times)^c \cap \mathbf{B}_j^\circ\big) \cap \mathbf{B}_{k,1}^\times\Big)
$$

$$
\leq 2\Big(H(\delta/\eta)\hat{t}(\epsilon)/\eta\Big)^{l^*-1} \cdot \mathbb{P}_x\Big(\mathbf{B}_{k,2}^\times \ \Big| \ \big(\bigcap_{j=1}^{k-1}(\mathbf{A}_j^\times \cup \mathbf{B}_j^\times)^c \cap \mathbf{B}_j^\circ\big) \cap \mathbf{B}_{k,1}^\times\Big)
$$

(for all $\eta$ sufficiently small due to Lemma G.4)

$$
\leq 4\Big(\frac{\hat{t}(\epsilon)}{\delta^\alpha}\Big)^{l^*-1}\Big(\frac{H(1/\eta)}{\eta}\Big)^{l^*-1} \cdot \mathbb{P}_x\Big(\mathbf{B}_{k,2}^\times \ \Big| \ \big(\bigcap_{j=1}^{k-1}(\mathbf{A}_j^\times \cup \mathbf{B}_j^\times)^c \cap \mathbf{B}_j^\circ\big) \cap \mathbf{B}_{k,1}^\times\Big)
$$

for all sufficiently small $\eta$, due to $H \in \mathcal{RV}_{-\alpha}(\eta)$. Meanwhile, note that

- $\big(\bigcap_{j=1}^{k-1}(\mathbf{A}_j^\times \cup \mathbf{B}_j^\times)^c \cap \mathbf{B}_j^\circ\big) \cap \mathbf{B}_{k,1}^\times \in \mathcal{F}_{T_{k,l^*}}$;

- on this event $\big(\bigcap_{j=1}^{k-1}(\mathbf{A}_j^\times \cup \mathbf{B}_j^\times)^c \cap \mathbf{B}_j^\circ\big) \cap \mathbf{B}_{k,1}^\times$ we have $\sigma(\eta) \wedge \widetilde{\tau}_k > T_{k,l^*}$ and $X_{T_{k,l^*}}^\eta \in [s_- + \epsilon, s_+ - \epsilon]$.

Therefore, using Lemma G.13 and strong Markov property again (at stopping time $T_{k,l^*}$), we know the following inequality holds for all $\eta$ sufficiently small:

$$
\sup_{k \geq 1} \sup_{|x| \leq 2\epsilon} \mathbb{P}_x\Big(\mathbf{B}_{k,2}^\times \ \Big| \ \big(\bigcap_{j=1}^{k-1}(\mathbf{A}_j^\times \cup \mathbf{B}_j^\times)^c \cap \mathbf{B}_j^\circ\big) \cap \mathbf{B}_{k,1}^\times\Big)
$$

$$
= \sup_{k \geq 1} \sup_{|x| \leq 2\epsilon} \mathbb{P}_x\Big(\big\{\sigma(\eta) > \widetilde{\tau}_k, \ \widetilde{\tau}_k - T_{k,l^*} \leq \rho(\epsilon)/\eta\big\}^c \ \Big| \ \big(\bigcap_{j=1}^{k-1}(\mathbf{A}_j^\times \cup \mathbf{B}_j^\times)^c \cap \mathbf{B}_j^\circ\big) \cap \mathbf{B}_{k,1}^\times\Big)
$$

$$
\leq \Psi(\epsilon)\frac{\delta^\alpha}{4\big(\hat{t}(\epsilon)/\delta^\alpha\big)^{l^*-1}}.
$$

Therefore, we know the existence of some $\eta_6 > 0$ such that

$$
\sup_{|x| \leq 2\epsilon} \mathbb{P}_x\Big(\mathbf{B}_k^\times \mid \bigcap_{j=1}^{k-1}(\mathbf{A}_j^\times \cup \mathbf{B}_j^\times)^c \cap \mathbf{B}_j^\circ\Big) \leq \Psi(\epsilon)\delta^\alpha\Big(\frac{H(1/\eta)}{\eta}\Big)^{l^*-1} \quad \forall \eta \in (0, \eta_6), \ \forall k \geq 1.
$$

$$
\tag{G.104}
$$

Similarly, we can bound conditional probabilities of the form $\mathbb{P}_x\big(\mathbf{A}_k^\times \mid \bigcap_{j=1}^{k-1}(\mathbf{A}_j^\times \cup \mathbf{B}_j^\times)^c \cap \mathbf{B}_j^\circ\big)$. To be specific, recall that $\mathbf{A}_k^\times = \cup_{i=0}^5 \mathbf{A}_{k,i}^\times$. By combining eq. (G.98)-eq. (G.103) with Markov property, we know the existence of some $\eta_7 > 0$ such that

$$
\sup_{|x| \leq 2\epsilon} \mathbb{P}_x\Big(\mathbf{A}_k^\times \mid \bigcap_{j=1}^{k-1}(\mathbf{A}_j^\times \cup \mathbf{B}_j^\times)^c \cap \mathbf{B}_j^\circ\Big) \leq 5\eta^N + 2\Psi(\epsilon)\delta^\alpha\Big(\frac{H(1/\eta)}{\eta}\Big)^{l^*-1} \quad \forall \eta \in (0, \eta_7), \ \forall k \geq 1.
$$

$$
\tag{G.105}
$$

On the other hand, a lower bound can be established for conditional probability involving $\mathbf{A}_k^\circ$, the event defined in eq. (G.94) describing the exit from $\Omega$ during an attempt with exactly $l^*$ large noises.

Using Lemma G.22 and Markov property of $(X_n^\eta)_{n\geq 1}$, one can see the existence of some $\eta_8 > 0$ such that

$$\inf_{|x|\leq 2\epsilon} \mathbb{P}_x\big(\mathbf{A}_k^\circ \mid \bigcap_{j=1}^{k-1}(\mathbf{A}_j^\times \cup \mathbf{B}_j^\times)^c \cap \mathbf{B}_j^\circ\big) \geq c_* \delta^\alpha \Big(\frac{H(1/\eta)}{\eta}\Big)^{l^*-1} \quad \forall \eta \in (0, \eta_8). \tag{G.106}$$

In order to apply the bounds eq. (G.104)-eq. (G.106), we make use of the following inclusion relationship:

$$\big(\bigcap_{j=1}^{k-1}(\mathbf{A}_j^\times \cup \mathbf{B}_j^\times)^c \cap \mathbf{B}_j^\circ\big) \cap (\mathbf{A}_k^\times \cup \mathbf{B}_k^\times)^c \subseteq \mathbf{A}_k^\circ \cup \mathbf{B}_k^\circ. \tag{G.107}$$

To see why this is true, let us consider a decomposition of the event on the L.H.S. of eq. (G.107). As mentioned above, on event $\bigcap_{j=1}^{k-1}(\mathbf{A}_j^\times \cup \mathbf{B}_j^\times)^c \cap \mathbf{B}_j^\circ$ we know that $\sigma(\eta) > \widetilde{\tau}_{k-1}$, so the $k$−th attempt occurred and there are only three possibilities on this event:

- $\mathbf{j}_k < l^*$;

- $\mathbf{j}_k \geq l^*$, $X_{T_{k,l^*}}^\eta \notin \Omega$;

- $\mathbf{j}_k \geq l^*$, $X_{T_{k,l^*}}^\eta \in \Omega$.

Let us partition the said event accordingly and analyze them one by one.

- On $\big(\bigcap_{j=1}^{k-1}(\mathbf{A}_j^\times \cup \mathbf{B}_j^\times)^c \cap \mathbf{B}_j^\circ\big) \cap (\mathbf{A}_k^\times \cup \mathbf{B}_k^\times)^c \cap \{\mathbf{j}_k < l^*\}$, due to the exclusion of $\mathbf{A}_k^\times$ (especially $\mathbf{A}_{k,1}^\times$ and $\mathbf{A}_{k,3}^\times$), we can see that if $\mathbf{j}_k < l^*$, then we must have $\sigma(\eta) > \widetilde{\tau}_k$ and $\widetilde{\tau}_k - T_{k,1} \leq 2l^*\hat{t}(\epsilon)/\eta$. Therefore,

$$\big(\bigcap_{j=1}^{k-1}(\mathbf{A}_j^\times \cup \mathbf{B}_j^\times)^c \cap \mathbf{B}_j^\circ\big) \cap (\mathbf{A}_k^\times \cup \mathbf{B}_k^\times)^c \cap \{\mathbf{j}_k < l^*\} \subseteq \mathbf{B}_k^\circ.$$

- On $\big(\bigcap_{j=1}^{k-1}(\mathbf{A}_j^\times \cup \mathbf{B}_j^\times)^c \cap \mathbf{B}_j^\circ\big) \cap (\mathbf{A}_k^\times \cup \mathbf{B}_k^\times)^c \cap \{\mathbf{j}_k \geq l^*, X_{T_{k,l^*}}^\eta \notin \Omega\}$, then the exclusion of $\mathbf{A}_{k,2}^\times$ implies that $T_{k,j} - T_{k,j-1} \leq 2\hat{t}(\epsilon)/\eta$ for all $j = 2, \cdots, l^*$, and the exclusion of $\mathbf{A}_{k,5}^\times$ tells us that if $X_{T_{k,l^*}}^\eta \notin \Omega$, then we have $X_{T_{k,l^*}}^\eta \notin [s_- - \epsilon, s_+ + \epsilon]$. In summary,

$$\big(\bigcap_{j=1}^{k-1}(\mathbf{A}_j^\times \cup \mathbf{B}_j^\times)^c \cap \mathbf{B}_j^\circ\big) \cap (\mathbf{A}_k^\times \cup \mathbf{B}_k^\times)^c \cap \{\mathbf{j}_k \geq l^*, X_{T_{k,l^*}}^\eta \notin \Omega\} \subseteq \mathbf{A}_k^\circ.$$

- On $\big(\bigcap_{j=1}^{k-1}(\mathbf{A}_j^\times \cup \mathbf{B}_j^\times)^c \cap \mathbf{B}_j^\circ\big) \cap (\mathbf{A}_k^\times \cup \mathbf{B}_k^\times)^c \cap \{\mathbf{j}_k \geq l^*, X_{T_{k,l^*}}^\eta \in \Omega\}$, the exclusion of $\mathbf{A}_{k,2}^\times$ again implies that $T_{k,j} - T_{k,j-1} \leq 2\hat{t}(\epsilon)/\eta$ for all $j = 2, \cdots, l^*$, hence $T_{k,l^*} - T_{k,1} \leq 2l^*\hat{t}(\epsilon)/\eta$. Similarly, the exclusion of $\mathbf{A}_{k,5}^\times$ tells us that if $X_{T_{k,l^*}}^\eta \in \Omega$, then we have $X_{T_{k,l^*}}^\eta \in [s_- + \epsilon, s_+ - \epsilon]$. Now since $\mathbf{B}_k^\times$ did not occur (see the definition in eq. (G.93)), we must have $\sigma(\eta) > \widetilde{\tau}_k$ and $\widetilde{\tau}_k - T_{k,l^*} \leq \rho(\epsilon)/\eta$, hence $\widetilde{\tau}_k - T_{k,1} \leq \frac{2l^*\hat{t}(\epsilon)+\rho(\epsilon)}{\eta}$. Therefore,

$$\big(\bigcap_{j=1}^{k-1}(\mathbf{A}_j^\times \cup \mathbf{B}_j^\times)^c \cap \mathbf{B}_j^\circ\big) \cap (\mathbf{A}_k^\times \cup \mathbf{B}_k^\times)^c \cap \{\mathbf{j}_k \geq l^*, X_{T_{k,l^*}}^\eta \in \Omega\} \subseteq \mathbf{B}_k^\circ.$$

Collecting results above, we have eq. (G.107). Now we discuss some of its implications. First, from eq. (G.107) we can immediately get that

$$\big(\bigcap_{j=1}^{k-1}(\mathbf{A}_j^\times \cup \mathbf{B}_j^\times)^c \cap \mathbf{B}_j^\circ\big) \cap (\mathbf{A}_k^\times \cup \mathbf{B}_k^\times)^c = \big(\bigcap_{j=1}^{k-1}(\mathbf{A}_j^\times \cup \mathbf{B}_j^\times)^c \cap \mathbf{B}_j^\circ\big) \cap (\mathbf{A}_k^\times \cup \mathbf{B}_k^\times)^c \cap (\mathbf{A}_k^\circ \cup \mathbf{B}_k^\circ).$$

$$\tag{G.108}$$

Next, recall the definitions of $\mathbf{A}_k^\circ$ in eq. (G.94) and $\mathbf{B}_k^\circ$ in eq. (G.95), and one can see that $\mathbf{A}_k^\circ$ and $\mathbf{B}_k^\circ$ are mutually exclusive, since the former implies that the first exit occurs during the $k$−th attempt while the latter implies that this attempt fails. This fact and eq. (G.108) allow us to conclude that

$$\bigcap_{i=1}^{k} \left(\mathbf{A}_i^\times \cup \mathbf{B}_i^\times \cup \mathbf{A}_i^\circ\right)^c = \bigcap_{i=1}^{k} \left(\mathbf{A}_i^\times \cup \mathbf{B}_i^\times\right)^c \cap \mathbf{B}_i^\circ = \left(\bigcap_{i=1}^{k-1} \left(\mathbf{A}_i^\times \cup \mathbf{B}_i^\times\right)^c \cap \mathbf{B}_i^\circ\right) \cap \left(\mathbf{A}_k^\times \cup \mathbf{B}_k^\times \cup \mathbf{A}_k^\circ\right)^c.$$
$$\text{(G.109)}$$

Now we use the results obtained so far to bound the probability of

$$\mathbf{A}^\times(\epsilon, \delta, \eta) \triangleq \bigcup_{k \geq 1} \left(\bigcap_{i=1}^{k-1} \left(\mathbf{A}_i^\times \cup \mathbf{B}_i^\times \cup \mathbf{A}_i^\circ\right)^c\right) \cap \left(\mathbf{A}_k^\times \cup \mathbf{B}_k^\times\right).$$

Using eq. (G.109), we can see that (for any $x \in [-2\epsilon, 2\epsilon]$)

$$
\begin{aligned}
&\mathbb{P}_x(\mathbf{A}^\times(\epsilon, \delta, \eta)) \\
&= \sum_{k \geq 1} \mathbb{P}_x\left(\left(\bigcap_{i=1}^{k-1} \left(\mathbf{A}_i^\times \cup \mathbf{B}_i^\times \cup \mathbf{A}_i^\circ\right)^c\right) \cap \left(\mathbf{A}_k^\times \cup \mathbf{B}_k^\times\right)\right) \\
&= \sum_{k \geq 1} \mathbb{P}_x\left(\left(\bigcap_{i=1}^{k-1} \left(\mathbf{A}_i^\times \cup \mathbf{B}_i^\times\right)^c \cap \mathbf{B}_i^\circ\right) \cap \left(\mathbf{A}_k^\times \cup \mathbf{B}_k^\times\right)\right) \\
&= \sum_{k \geq 1} \mathbb{P}_x\left(\mathbf{A}_k^\times \cup \mathbf{B}_k^\times \,\Big|\, \bigcap_{i=1}^{k-1} \left(\mathbf{A}_i^\times \cup \mathbf{B}_i^\times\right)^c \cap \mathbf{B}_i^\circ\right) \\
&\qquad \cdot \prod_{j=1}^{k-1} \mathbb{P}_x\left(\bigcap_{i=1}^{j} \left(\mathbf{A}_i^\times \cup \mathbf{B}_i^\times\right)^c \cap \mathbf{B}_i^\circ \,\Big|\, \bigcap_{i=1}^{j-1} \left(\mathbf{A}_i^\times \cup \mathbf{B}_i^\times\right)^c \cap \mathbf{B}_i^\circ\right) \\
&= \sum_{k \geq 1} \mathbb{P}_x\left(\mathbf{A}_k^\times \cup \mathbf{B}_k^\times \,\Big|\, \bigcap_{i=1}^{k-1} \left(\mathbf{A}_i^\times \cup \mathbf{B}_i^\times\right)^c \cap \mathbf{B}_i^\circ\right) \\
&\qquad \cdot \prod_{j=1}^{k-1} \mathbb{P}_x\left(\left(\bigcap_{i=1}^{j-1} \left(\mathbf{A}_i^\times \cup \mathbf{B}_i^\times\right)^c \cap \mathbf{B}_i^\circ\right) \cap \left(\mathbf{A}_j^\times \cup \mathbf{B}_j^\times \cup \mathbf{A}_j^\circ\right)^c \,\Big|\, \bigcap_{i=1}^{j-1} \left(\mathbf{A}_i^\times \cup \mathbf{B}_i^\times\right)^c \cap \mathbf{B}_i^\circ\right) \\
&= \sum_{k \geq 1} \mathbb{P}_x\left(\mathbf{A}_k^\times \cup \mathbf{B}_k^\times \,\Big|\, \bigcap_{i=1}^{k-1} \left(\mathbf{A}_i^\times \cup \mathbf{B}_i^\times\right)^c \cap \mathbf{B}_i^\circ\right) \\
&\qquad \cdot \prod_{j=1}^{k-1} \mathbb{P}_x\left(\left(\mathbf{A}_j^\times \cup \mathbf{B}_j^\times \cup \mathbf{A}_j^\circ\right)^c \,\Big|\, \bigcap_{i=1}^{j-1} \left(\mathbf{A}_i^\times \cup \mathbf{B}_i^\times\right)^c \cap \mathbf{B}_i^\circ\right) \\
&\leq \sum_{k \geq 1} \mathbb{P}_x\left(\mathbf{A}_k^\times \cup \mathbf{B}_k^\times \,\Big|\, \bigcap_{i=1}^{k-1} \left(\mathbf{A}_i^\times \cup \mathbf{B}_i^\times\right)^c \cap \mathbf{B}_i^\circ\right) \cdot \prod_{j=1}^{k-1} \left(1 - \mathbb{P}_x\left(\mathbf{A}_j^\circ \,\Big|\, \bigcap_{i=1}^{j-1} \left(\mathbf{A}_i^\times \cup \mathbf{B}_i^\times\right)^c \cap \mathbf{B}_i^\circ\right)\right).
\end{aligned}
$$

This allows us to apply eq. (G.104)-eq. (G.106) and conclude that (here we only consider $\eta < \min\{\eta_i : i \in [8]\}$ ),

$$\sup_{|x|\le 2\epsilon} \mathbb{P}_x(\mathbf{A}^\times(\epsilon,\delta,\eta))$$

$$\le \sum_{k\ge 1} \left(5\eta^N + 2\Psi(\epsilon)\delta^\alpha \left(\frac{H(1/\eta)}{\eta}\right)^{l^*-1}\right) \cdot \left(1 - c_*\delta^\alpha\left(\frac{H(1/\eta)}{\eta}\right)^{l^*-1}\right)^{k-1}$$

$$= \frac{5\eta^N + 2\Psi(\epsilon)\delta^\alpha\left(\frac{H(1/\eta)}{\eta}\right)^{l^*-1}}{c_*\delta^\alpha\left(\frac{H(1/\eta)}{\eta}\right)^{l^*-1}}$$

$$\le \frac{2\Psi(\epsilon) + 5\eta^\alpha}{c_*} \quad \text{for sufficiently small } \eta, \text{ due to } H \in \mathcal{RV}_{-\alpha}(\eta) \text{ and our choice of } N > \alpha l^*$$

$$\le \frac{3\Psi(\epsilon)}{c_*} < C \quad \text{for all } \eta \text{ small enough such that } 5\eta^\alpha < \Psi(\epsilon).$$

The last inequality follows from our choice of $\epsilon$ in eq. (G.97). This concludes the proof. $\qquad\square$

Having established Lemma G.24, we return to Proposition G.23 and give a proof. Recall that, aside from the attraction field $\Omega = (s_-, s_+)$, there are $n_{\min} - 1$ other attraction fields $\widetilde{\Omega}_k = (s_k^-, s_k^+)$ (for each $k \in [n_{\min} - 1]$). Besides, the function $\lambda(\cdot)$ and constants $\nu^\Omega, \nu_k^\Omega$ are defined in eq. (G.77)-eq. (G.79).

*Proof of Proposition G.23.* We fix some parameters for the proof. First, with out loss of generality we only need to consider $C \in (0, 1)$. Next we discuss the valid range of $\epsilon$ for the claim to hold. We only consider $\epsilon > 0$ such that

$$\epsilon < \frac{\bar{\epsilon}}{6(\bar{\rho} + \widetilde{\rho} + 3)} \wedge \frac{\epsilon_0}{3}$$

where $\bar{\rho}$ and $\widetilde{\rho}$ are the constants in Corollary G.7 and Corollary G.10 respectively, and $\epsilon_0$ is the constant in eq. (G.8). Due to continuity of measure $\mu$, it holds for all $\epsilon$ small enough such that (let $\hat{\epsilon} = 3(\bar{\rho} + \widetilde{\rho} + 3)\epsilon$)

$$\frac{\mu(E(0))}{\mu(E(\hat{\epsilon}))} < 1/(1 - C), \qquad \text{(G.110)}$$

$$\frac{\mu(E(0))}{\mu(E(-\hat{\epsilon}))} > 1/(1 + C), \qquad \text{(G.111)}$$

$$\frac{\mu\left(h^{-1}\left((s_- - 2\hat{\epsilon}, s_- + 2\hat{\epsilon}) \cup (s_+ - 2\hat{\epsilon}, s_+ + 2\hat{\epsilon})\right)\right)}{\mu(E(-\hat{\epsilon}))} \le C \qquad \text{(G.112)}$$

$$\frac{\mu\left(E(\hat{\epsilon}) \cap (s_{k'}^- - \hat{\epsilon}, s_{k'}^+ + \hat{\epsilon})\right)}{\mu(E(\hat{\epsilon}))} \le \frac{\nu_{k'}^\Omega + C}{\nu^\Omega} \qquad \text{(G.113)}$$

$$\frac{\mu\left(E(-\hat{\epsilon}) \cap (s_{k'}^- + 2\hat{\epsilon}, s_{k'}^+ - 2\hat{\epsilon})\right)}{\mu(E(-\hat{\epsilon}))} \ge \frac{\nu_{k'}^\Omega - C}{\nu^\Omega} \qquad \text{(G.114)}$$

In our proof we only consider $\epsilon$ small enough so the inequality above holds, and the claims in Lemma G.13 hold. Moreover, we only consider $\epsilon$ and $\delta$ small enough so that Lemma G.24 hold and we have

$$\lim_{\eta \downarrow 0} \sup_{|x| \le 2\epsilon} \mathbb{P}_x(\mathbf{A}^\times(\epsilon,\delta,\eta)) < C. \qquad \text{(G.115)}$$

We show that the desired claims hold for all $\epsilon, \delta$ sufficiently small that satisfy conditions above.

First, in order to show eq. (G.84), we define event

$$\widetilde{\mathbf{A}}^\times(\epsilon,\delta,\eta) \triangleq \left(\mathbf{A}^\times(\epsilon,\delta,\eta)\right)^c \cap$$

$$\left\{\lambda(\eta)\left(\tau(\eta,\epsilon) - \sigma(\eta)\right) \ge C \text{ or } \exists n = \sigma(\eta) + 1, \cdots, \tau(\eta,\epsilon) \text{ such that } X_n^\eta \notin \widetilde{\Omega}_{J_{\sigma(\eta)}}\right\}.$$

Since $\lambda \in \mathcal{RV}_{-1-l^*(\alpha-1)}$ and $\alpha > 1$, for the $\epsilon$ we fixed at the beginning of this proof, $\rho(\epsilon)$ is a fixed constant as well (the function $\rho$ is defined in Lemma G.13) and we have $\lim_{\eta\downarrow 0} \lambda(\eta)\rho(\epsilon)/\eta = 0$. Next, the occurrence of $\left(\mathbf{A}^\times(\epsilon, \delta, \eta)\right)^c$ (in particular, the exclusion of all the $\mathbf{A}^\times_{k,5}$ defined in eq. (G.91)), we know that $X^\eta_{\sigma(\eta)} \notin [s^-_{J_\sigma} - \epsilon, s^-_{J_\sigma} + \epsilon] \cup [s^+_{J_\sigma} - \epsilon, s^+_{J_\sigma} + \epsilon]$ (recall that for any $k \in [n_{\min} - 1]$, we have $\widetilde{\Omega}_j = (s^-_j, s^+_j)$; for definition of $J_\sigma$ see eq. (G.76)). Meanwhile, for all $\eta$ sufficiently small, we have $\epsilon/\lambda(\eta) > \rho(\epsilon)/\eta$. Therefore, using Lemma G.13 we can see that (for all $\eta$ sufficiently small)

$$\sup_{|x| \leq 2\epsilon} \mathbb{P}_x\left(\widetilde{\mathbf{A}}^\times(\epsilon, \delta, \eta) \mid \left(\mathbf{A}^\times(\epsilon, \delta, \eta)\right)^c\right) \leq C. \tag{G.116}$$

Lastly, observe that

$$\mathbb{P}\left(\left\{\lambda(\eta)\left(\tau(\eta, \epsilon) - \sigma(\eta)\right) \geq C \text{ or } \exists n = \sigma(\eta) + 1, \cdots, \tau(\eta, \epsilon) \text{ such that } X^\eta_n \notin \widetilde{\Omega}_{J_\sigma(\eta)}\right\}\right)$$

$$\leq \mathbb{P}_x\left((\mathbf{A}^\times)^c \cap \left\{\lambda(\eta)\left(\tau(\eta, \epsilon) - \sigma(\eta)\right) \geq C\right.\right.$$

$$\left.\left. \text{or } \exists n = \sigma(\eta) + 1, \cdots, \tau(\eta, \epsilon) \text{ such that } X^\eta_n \notin \widetilde{\Omega}_{J_\sigma(\eta)}\right\}\right) + \mathbb{P}_x(\mathbf{A}^\times)$$

so by combining eq. (G.115) with eq. (G.116), we can obtain eq. (G.84).

Moving on, we discuss the upper bounds eq. (G.80) and eq. (G.82). Recall that the fixed constant $k' \in [n_{\min} - 1]$ is prescribed in the description of this proposition. Let us observe some facts on event $(\mathbf{A}^\times(\epsilon, \eta, \delta))^c \cap \{J_\sigma(\eta) = k'\}$: If we let $J(\epsilon, \delta, \eta) \triangleq \sup\{k \geq 0 : \widetilde{\tau}_k < \sigma(\eta)\}$ be the number of attempts it took to escape, and

$$J^\uparrow(\epsilon, \delta, \eta) \triangleq \min\{k \geq 1 : T_{k,1} \text{ has } \left(3(\bar{\rho} + \widetilde{\rho} + 3)\epsilon, \delta, \eta\right) - \text{overflow}\},$$

then for all $\eta$ sufficiently small, we must have $J \leq J^\uparrow$ on event $(\mathbf{A}^\times(\epsilon, \eta, \delta))^c \cap \{J_\sigma(\eta) = k'\}$. To see this via a proof of contradiction, let us assume that, for some arbitrary positive integer $j$, there exists some sample path on $(\mathbf{A}^\times)^c \cap \{J_\sigma(\eta) = k'\}$ such that $J^\uparrow = j < J$. Then from the definition of $(\mathbf{A}^\times)^c$, in particular the exclusion of event $\mathbf{A}^\times_{j,0}$ (see the definition in eq. (G.86)), for all sufficiently small $\eta$, we are able to apply Corollary G.10 and G.7 and conclude that $X^\eta_{T_{j,l^*}} \notin \Omega$: indeed, using Corollary G.10 and G.7 we can show that the distance between $X^\eta_{T_{j,l^*}}$ and the perturbed ODE

$$\widetilde{\mathbf{x}}^\eta\left(T_{j,l^*} - T_{j,1}, 0; \left(0, T_{j,2} - T_{j,1}, \cdots, T_{j,l^*} - T_{j,1}\right), \left(\eta W_{j,1}, \cdots, \eta W_{j,l^*}\right)\right)$$

is strictly less than $3(\bar{\rho} + \widetilde{\rho} + 3)\epsilon$; on the other hand, the definition of $\left(3(\bar{\rho} + \widetilde{\rho} + 3)\epsilon, \delta, \eta\right) - \text{overflow}$ implies that

$$\widetilde{\mathbf{x}}^\eta\left(T_{j,l^*} - T_{j,1}, 0; \left(0, T_{j,2} - T_{j,1}, \cdots, T_{j,l^*} - T_{j,1}\right), \left(\eta W_{j,1}, \cdots, \eta W_{j,l^*}\right)\right)$$

$$\notin [s_- - 3(\bar{\rho} + \widetilde{\rho} + 3)\epsilon, s_+ + 3(\bar{\rho} + \widetilde{\rho} + 3)\epsilon].$$

Therefore, we must have $X^\eta_{T_{j,l^*}} \notin \Omega$, which contradicts our assumption $j = J^\uparrow < J$. In summary, we have shown that, on $(\mathbf{A}^\times)^c \cap \{J_\sigma = k'\}$, we have $J^\uparrow(\epsilon, \delta, \eta) \geq J(\epsilon, \delta, \eta)$. Similarly, if we consider

$$J^\downarrow(\epsilon, \delta, \eta) \triangleq \min\{k \geq 1 : T_{k,1} \text{ has } \left(-3(\bar{\rho} + \widetilde{\rho} + 3)\epsilon, \delta, \eta\right) - \text{overflow}\},$$

then by the same argument above we can show that $J^\downarrow(\epsilon, \delta, \eta) \leq J(\epsilon, \delta, \eta)$. Now consider the following decomposition of events.

- On $\{J^\downarrow < J^\uparrow\}$, we know that for the first $k$ such that $T_{k,1}$ has $\left(-3(\bar{\rho} + \widetilde{\rho} + 3)\epsilon, \delta, \eta\right) - \text{overflow}$, it does not have $\left(3(\bar{\rho} + \widetilde{\rho} + 3)\epsilon, \delta, \eta\right) - \text{overflow}$. Now we analyze the probability that $Z_0$ does not have $(\hat{\epsilon}, \delta, \eta) - \text{overflow}$ conditioning on that it does have $(-\hat{\epsilon}, \delta, \eta) - \text{overflow}$ (recall that we let $\hat{\epsilon} = 3(\bar{\rho} + \widetilde{\rho} + 3)$). Using Lemma G.15 and the bound eq. (G.112), we know that for all $\eta$ sufficiently small,

$$\sup_{|x| \leq 2\epsilon} \mathbb{P}_x\left((\mathbf{A}^\times)^c \cap \{J^\downarrow < J^\uparrow\}\right)$$

$$\leq \frac{\mu\left(h^{-1}\left((s_- - 2\hat{\epsilon}, s_- + 2\hat{\epsilon}) \cup (s_+ - 2\hat{\epsilon}, s_+ + 2\hat{\epsilon})\right)\right)}{\mu(E(-\hat{\epsilon}))} \leq C. \tag{G.117}$$

- On $(\mathbf{A}^\times)^c \cap \{J_\sigma = k'\} \cap \{J^\uparrow = J^\downarrow\}$, due to $J^\uparrow = J^\downarrow = J$ we know that $T_{J(\epsilon,\delta,\eta),1}$ is the first among all $T_{k,1}$ to have $(\hat{\epsilon}, \delta, \eta)-$overflow. Moreover, due to $\{J_\sigma = k'\}$ and using Corollary G.10 and G.7 again as we did above, we know that the overflow endpoint of $T_{J(\epsilon,\delta,\eta),1}$ is in $(s_{k'}^- - \hat{\epsilon}, s_{k'}^+ + \hat{\epsilon})$ (recall that $\widetilde{\Omega}_{k'} = (s_{k'}^-, s_{k'}^+)$). In summary, for any $n \geq 0$

$$(\mathbf{A}^\times)^c \cap \{J_\sigma = k'\} \cap \{J^\uparrow = J^\downarrow > n\}$$
$$\subseteq (\mathbf{A}^\times)^c \cap \{J^\uparrow > n\} \cap \left\{ T_{J^\uparrow,1} \text{ has overflow endpoint in } (s_{k'}^- - \hat{\epsilon}, s_{k'}^+ + \hat{\epsilon}) \right\}$$

so using Lemma G.15, we obtain that (for all $\eta$ sufficiently small)

$$\sup_{|x| \leq 2\epsilon} \mathbb{P}_x \left( (\mathbf{A}^\times)^c \cap \{J_\sigma = k'\} \cap \{J^\uparrow = J^\downarrow > n\} \right)$$
$$\leq \sup_{|x| \leq 2\epsilon} \mathbb{P}_x \left( (\mathbf{A}^\times)^c \cap \{J^\uparrow > n\} \right) \cdot \frac{p(\hat{\epsilon}, \delta, \eta; (s_{k'}^- - \hat{\epsilon}, s_{k'}^+ + \hat{\epsilon}))}{p(\hat{\epsilon}, \delta, \eta)}$$
$$\leq \sup_{|x| \leq 2\epsilon} \mathbb{P}_x \left( (\mathbf{A}^\times)^c \cap \{J^\uparrow > n\} \right) \cdot \frac{\nu_{k'}^\Omega + C}{\nu^\Omega}. \tag{G.118}$$

uniformly for any $n = 0, 1, 2, \cdots$ due to eq. (G.113).

- On the other hand, on $(\mathbf{A}^\times)^c$, if $T_{J^\downarrow,1}$ has overflow endpoint in $(s_{k'}^- + 2\hat{\epsilon}, s_{k'}^+ - 2\hat{\epsilon})$, then from Definition G.1 we know that $T_{J^\downarrow,1}$ also has $(\hat{\epsilon}, \delta, \eta)-$overflow, hence $J^\downarrow = J^\uparrow = J$. Moreover, using Corollary G.10 and G.7 again, we know that $X_{T_{J^\downarrow,l^*}}^\eta \in (s_{k'}^-, s_{k'}^+)$ so $J_\sigma = k'$. In summary, for any $n \geq 0$,

$$(\mathbf{A}^\times)^c \cap \{J_\sigma = k'\} \cap \{J^\uparrow = J^\downarrow > n\}$$
$$\supseteq (\mathbf{A}^\times)^c \cap \{J^\downarrow > n\} \cap \left\{ T_{J^\downarrow,1} \text{ has overflow endpoint in } (s_{k'}^- + 2\hat{\epsilon}, s_{k'}^+ - 2\hat{\epsilon}) \right\}$$

so using Lemma G.15, we obtain that (for all $\eta$ sufficiently small)

$$\inf_{|x| \leq 2\epsilon} \mathbb{P}_x \left( (\mathbf{A}^\times)^c \cap \{J_\sigma = k'\} \cap \{J^\uparrow = J^\downarrow > n\} \right)$$
$$\geq \inf_{|x| \leq 2\epsilon} \mathbb{P}_x \left( (\mathbf{A}^\times)^c \cap \{J^\downarrow > n\} \right) \cdot \frac{p\left( -\hat{\epsilon}, \delta, \eta; (s_{k'}^- + 2\hat{\epsilon}, s_{k'}^+ - 2\hat{\epsilon}) \right)}{p(-\hat{\epsilon}, \delta, \eta)}$$
$$\geq \inf_{|x| \leq 2\epsilon} \mathbb{P}_x \left( (\mathbf{A}^\times)^c \cap \{J^\downarrow > n\} \right) \cdot \frac{\nu_{k'}^\Omega - C}{\nu^\Omega}. \tag{G.119}$$

uniformly for any $n = 0, 1, 2, \cdots$ due to eq. (G.114).

Besides, the following claim holds on event $(\mathbf{A}^\times)^c$.

- From eq. (G.109), the definition of $\mathbf{B}_k^\circ$ as well as the definition of event $\mathbf{A}_k^\circ$ (see eq. (G.94)), one can see that for any $j = 1, 2, \cdots, J$, we have

$$\widetilde{\tau}_j \wedge \sigma(\eta) - T_{j,1} \leq \frac{2l^* \hat{t}(\epsilon) + \rho(\epsilon)}{\eta}.$$

- Now if we turn to the interval $(\widetilde{\tau}_{j-1}, T_{j,1}]$ (the time between the start of the $j-$th attempt and the arrival of the first large noise during this attempt) for each $j = 1, 2, \cdots, J$, and the following sequence constructed by concatenating these intervals

$$\mathbf{S}(\epsilon, \delta, \eta) \triangleq \left( 1, 2, \cdots, T_{1,1}, \widetilde{\tau}_1 + 1, \widetilde{\tau}_1 + 2, \cdots, T_{2,1}, \cdots, \right.$$
$$\left. \widetilde{\tau}_k + 1, \widetilde{\tau}_k + 2, \cdots, T_{k+1,1}, \widetilde{\tau}_{k+1} + 1, \widetilde{\tau}_{k+2} + 1, \cdots \right),$$

then the discussion above have shown that, for

$$\min\{n \in \mathbf{S}(\epsilon, \delta, \eta): \ Z_n \text{ has } \left( 3(\bar{\rho} + \widetilde{\rho} + 3)\epsilon, \delta, \eta \right)-\text{overflow}\} \geq T_{J,1}.$$

Meanwhile, from the definition of overflow we know that the probability that $Z_1$ has $\left(3(\bar{\rho} + \widetilde{\rho} + 3)\epsilon, \delta, \eta\right)$−overflow is equal to

$$H(\delta/\eta)p\Big(3(\bar{\rho} + \widetilde{\rho} + 3)\epsilon, \delta, \eta\Big).$$

- Therefore, if, within the duration of each attempt, we split the attempt into two parts at the arrival time of the first large jump $(T_{k,1})_{k \geq 1}$ at each attempt, and define (here the subscript *before* or *after* indicates that we are counting the steps before or after the first large jump in an attempt)

$$\mathbf{S}_{\text{before}}(\epsilon, \delta, \eta) \triangleq \{n \in \mathbf{S}(\epsilon, \delta, \eta): \ n \leq \sigma(\eta)\}, \ I_{\text{before}}(\epsilon, \delta, \eta) \triangleq \#\mathbf{S}_{\text{before}}(\epsilon, \delta, \eta),$$
$$\mathbf{S}_{\text{after}}(\epsilon, \delta, \eta) \triangleq \{n \notin \mathbf{S}(\epsilon, \delta, \eta): \ n \leq \sigma(\eta)\}, \ I_{\text{after}}(\epsilon, \delta, \eta) \triangleq \#\mathbf{S}_{\text{after}}(\epsilon, \delta, \eta),$$

then we have $\sigma(\eta) = I_{\text{before}} + I_{\text{after}}$. Moreover, the discussion above implies that

$$I_{\text{after}} \leq J\big(2l^*\hat{t}(\epsilon) + \rho(\epsilon)\big)/\eta$$
$$I_{\text{before}} \leq \min\{n \in \mathbf{S}(\epsilon, \delta, \eta): \ Z_n \ \text{has} \ \Big(3(\bar{\rho} + \widetilde{\rho} + 3)\epsilon, \delta, \eta\Big)-\text{overflow}\}$$

and on event $(\mathbf{A}^\times)^c$.

Define geometric random variables with the following success rates

$$U_1(\epsilon, \delta, \eta) \sim \text{Geom}\Big(p\big(3(\bar{\rho} + \widetilde{\rho} + 3)\epsilon, \delta, \eta\big)\Big)$$
$$U_2(\epsilon, \delta, \eta) \sim \text{Geom}\Big(H(\delta/\eta)p\big(3(\bar{\rho} + \widetilde{\rho} + 3)\epsilon, \delta, \eta\big)\Big).$$

Using results above to bound $I_{\text{before}}$ and $I_{\text{after}}$ separately on event $(\mathbf{A}^\times)^c$, we can show that (for all $\eta$ sufficiently small and any $u > 0$)

$$\sup_{x \in [-2\epsilon, 2\epsilon]} \mathbb{P}_x\Big(v^\Omega \lambda(\eta)\sigma(\eta) > u, J_\sigma(\eta) = k'\Big)$$

$$\leq \sup_{|x| \leq 2\epsilon} \mathbb{P}_x(\mathbf{A}^\times(\epsilon, \delta, \eta)) + \sup_{x \in [-2\epsilon, 2\epsilon]} \mathbb{P}_x\Big(\{v^\Omega \lambda(\eta)\sigma(\eta) > u, J_\sigma(\eta) = k'\} \cap \big(\mathbf{A}^\times(\epsilon, \delta, \eta)\big)^c\Big)$$

$$\leq C + \sup_{x \in [-2\epsilon, 2\epsilon]} \mathbb{P}_x\Big(\{v^\Omega \lambda(\eta)\sigma(\eta) > u, J_\sigma(\eta) = k'\} \cap \big(\mathbf{A}^\times(\epsilon, \delta, \eta)\big)^c\Big) \quad \text{due to eq. (G.115)}$$

$$\leq C + \sup_{x \in [-2\epsilon, 2\epsilon]} \mathbb{P}_x\Big(\{v^\Omega \lambda(\eta)I_{\text{before}}(\epsilon, \delta, \eta) > (1 - C)u, J_\sigma(\eta) = k'\} \cap \big(\mathbf{A}^\times(\epsilon, \delta, \eta)\big)^c\Big)$$

$$+ \sup_{x \in [-2\epsilon, 2\epsilon]} \mathbb{P}_x\Big(\{v^\Omega \lambda(\eta)I_{\text{after}}(\epsilon, \delta, \eta) > Cu\} \cap \big(\mathbf{A}^\times(\epsilon, \delta, \eta)\big)^c\Big)$$

$$\leq C + \sup_{x \in [-2\epsilon, 2\epsilon]} \mathbb{P}_x\Big(\{v^\Omega \lambda(\eta)I_{\text{before}}(\epsilon, \delta, \eta) > (1 - C)u, J_\sigma(\eta) = k'\} \cap \big(\mathbf{A}^\times(\epsilon, \delta, \eta)\big)^c\Big)$$

$$+ \mathbb{P}\Big(v^\Omega \lambda(\eta)\frac{\rho(\epsilon) + 2l^*\hat{t}(\epsilon)}{\eta} \cdot U_1(\epsilon, \delta, \eta) > Cu\Big)$$

$$\leq C$$

$$+ \sup_{x \in [-2\epsilon, 2\epsilon]} \mathbb{P}_x\Big(\{v^\Omega \lambda(\eta)I_{\text{before}}(\epsilon, \delta, \eta) > (1 - C)u, J_\sigma(\eta) = k'\} \cap \big(\mathbf{A}^\times(\epsilon, \delta, \eta)\big)^c \cap \{J^\downarrow = J^\uparrow\}\Big)$$

$$+ \sup_{x \in [-2\epsilon, 2\epsilon]} \mathbb{P}_x\big((\mathbf{A}^\times)^c \cap \{J^\downarrow < J^\uparrow\}\big) + \mathbb{P}\Big(v^\Omega \lambda(\eta)\frac{\rho(\epsilon) + 2l^*\hat{t}(\epsilon)}{\eta} \cdot U_1(\epsilon, \delta, \eta) > Cu\Big)$$

$$\leq 2C + \mathbb{P}\Big(v^\Omega \lambda(\eta)U_2(\epsilon, \delta, \eta) > (1 - C)u\Big)\frac{\nu^\Omega_{k'} + C}{\nu^\Omega}$$

$$+ \mathbb{P}\Big(v^\Omega \lambda(\eta)\frac{\rho(\epsilon) + 2l^*\hat{t}(\epsilon)}{\eta} \cdot U_1(\epsilon, \delta, \eta) > Cu\Big) \tag{G.120}$$

where the last inequality follows from eq. (G.117) and eq. (G.118). Now let us analyze the probability terms on the last row of the display above. For the first term, let $a(\eta) = H(\delta/\eta)p\Big(3(\bar{\rho}+\widetilde{\rho}+3)\epsilon, \delta, \eta\Big)$. Due to Lemma G.15, we have (recall that $\nu^{\Omega} = \mu(E(0))$)

$$\lim_{\eta\downarrow 0} \frac{a(\eta)}{\lambda(\eta)\mu\big(E\big(3(\bar{\rho} + \widetilde{\rho} + 3)\epsilon\big)\big)} = 1.$$

Combining this with eq. (G.110), one can see that for all $\eta$ sufficiently small,

$$\mathbb{P}\Big(v^{\Omega}\lambda(\eta)U_2(\epsilon, \delta, \eta) > (1 - C)u\Big) \le \mathbb{P}\Big(a(\eta)\mathrm{Geom}\big(a(\eta)\big) > (1 - C)^2 u\Big) \ \forall u > 0.$$

Next, let $b(\eta, u) = \mathbb{P}\Big(a(\eta)\mathrm{Geom}\big(a(\eta)\big) > (1 - C)^2 u\Big) = \mathbb{P}\Big(\mathrm{Geom}\big(a(\eta)\big) > \frac{(1-C)^2 u}{a(\eta)}\Big)$. For $g(y) = \log(1 - y)$, we know the existence of some $y_0 > 0$ such that for all $y \in (0, y_0)$, we have $\log(1 - y) \le -(1 - C)y$. So one can see that for all $\eta$ sufficiently small,

$$\log b(u, \eta) \le \frac{(1 - C)^2 u}{a(\eta)} \log(1 - a(\eta)) \le -(1 - C)^3 u$$

$$\Rightarrow b(u, \eta) \le \exp\big(-(1 - C)^3 u\big) \tag{G.121}$$

uniformly for all $u > 0$.

For the second probability term, if we only consider $u \ge C$, then

$$\mathbb{P}\Big(v^{\Omega}\lambda(\eta)\frac{\rho(\epsilon) + 2l^*\hat{t}(\epsilon)}{\eta} \cdot U_1(\epsilon, \delta, \eta) > Cu\Big) \le \mathbb{P}\Big(v^{\Omega}\lambda(\eta)\frac{\rho(\epsilon) + 2l^*\hat{t}(\epsilon)}{\eta} \cdot U_1(\epsilon, \delta, \eta) > C^2\Big).$$

Using $H \in \mathcal{RV}_{-\alpha}(\eta)$ with $\alpha > 1$, we get

$$p\Big(3(\bar{\rho} + \widetilde{\rho} + 3)\epsilon, \delta, \eta\Big)U_1(\epsilon, \delta, \eta) \xrightarrow{d} \mathrm{Exp}(1) \quad \text{as } \eta \downarrow 0$$

due to the nature of the Geometric random variable $U_1$. Besides, due to $H \in \mathcal{RV}_{-\alpha}(\eta)$ with $\alpha > 1$ and Lemma G.15, it is easy to show that

$$\lim_{\eta\downarrow 0} \frac{\lambda(\eta)\frac{\rho(\epsilon) + 2l^*\hat{t}(\epsilon)}{\eta}}{p\Big(3(\bar{\rho} + \widetilde{\rho} + 3)\epsilon, \delta, \eta\Big)} = 0.$$

Combining these results with Slutsky's theorem, we now obtain

$$\mu(E(0))\lambda(\eta)\frac{\rho(\epsilon) + 2l^*\hat{t}(\epsilon)}{\eta} \cdot U_1(\epsilon, \delta, \eta) \xrightarrow{d} 0 \quad \text{as } \eta \downarrow 0.$$

Therefore,

$$\limsup_{\eta\downarrow 0} \sup_{u\ge C} \mathbb{P}\Big(\mu(E(0))\lambda(\eta)\frac{\rho(\epsilon) + 2l^*\hat{t}(\epsilon)}{\eta} \cdot U_1(\delta, \eta) > Cu\Big) = 0. \tag{G.122}$$

Plugging eq. (G.121) and eq. (G.122) back into eq. (G.120), we can establish the upper bound in eq. (G.80). To show eq. (G.82), note that for event

$$E(\epsilon, \eta) = \{\nu^{\Omega}\lambda(\eta)\tau(\eta, \epsilon) > u, \ X_{\tau(\eta,\epsilon)}^{\eta} \in B(\widetilde{m}_{k'}, 2\epsilon)\},$$

we have (for definitions of $\tau$, see eq. (G.75))

$$E(\epsilon, \eta) \supseteq \{v^{\Omega}\lambda(\eta)\sigma(\eta) > u, J_{\sigma}(\eta) = k'\} \cap \{X_n^{\eta} \in \widetilde{\Omega}_{J_{\sigma}(\eta)} \ \forall n \in [\sigma(\eta), \tau(\eta, \epsilon)]\},$$

$$E(\epsilon, \eta) \cap \{v^{\Omega}\lambda(\eta)\sigma(\eta) > u, J_{\sigma}(\eta) = j\} \cap \{X_n^{\eta} \in \widetilde{\Omega}_{J_{\sigma}(\eta)} \ \forall n \in [\sigma(\eta), \tau(\eta, \epsilon)]\} = \emptyset \ \forall j \ne k'.$$

Therefore, for all $\eta$ sufficiently small,

$$\sup_{|x|\le 2\epsilon} \mathbb{P}_x(E(\epsilon, \eta))$$

$$\le \sup_{|x|\le 2\epsilon} \mathbb{P}_x(\mathbf{A}^{\times}) + \sup_{|x|\le 2\epsilon} \mathbb{P}_x\big((\mathbf{A}^{\times})^c \cap \{X_n^{\eta} \notin \widetilde{\Omega}_{J_{\sigma}(\eta)} \text{ for some } n \in [\sigma(\eta), \tau(\eta, \epsilon)]\}\big)$$

$$+ \sup_{|x|\le 2\epsilon} \mathbb{P}_x\Big((\mathbf{A}^{\times})^c \cap \{v^{\Omega}\lambda(\eta)\sigma(\eta) > u, J_{\sigma}(\eta) = k'\} \cap \{X_n^{\eta} \in \widetilde{\Omega}_{J_{\sigma}(\eta)} \ \forall n \in [\sigma(\eta), \tau(\eta, \epsilon)]\}\Big)$$

$$\le 4C + \exp\big(-(1 - C)^3 u\big)\frac{\nu_{k'}^{\Omega} + C}{\nu^{\Omega}}$$

uniformly for all $u \geq C$, due to eq. (G.115), eq. (G.84) and eq. (G.120).

The lower bound can be shown by an almost identical approach. In particular, analogous to eq. (G.120), we can show that (for any $u > 0$)

$$\inf_{x \in [-2\epsilon, 2\epsilon]} \mathbb{P}_x \left( v^\Omega \lambda(\eta) \sigma(\eta) > u, J_\sigma(\eta) = k' \right)$$

$$\geq \inf_{x \in [-2\epsilon, 2\epsilon]} \mathbb{P}_x \left( \{ v^\Omega \lambda(\eta) \sigma(\eta) > u, J_\sigma(\eta) = k' \} \cap \left( \mathbf{A}^\times(\epsilon, \delta, \eta) \right)^c \right)$$

$$\geq \inf_{x \in [-2\epsilon, 2\epsilon]} \mathbb{P}_x \left( \{ v^\Omega \lambda(\eta) I_{\text{before}}(\epsilon, \delta, \eta) > u, J_\sigma(\eta) = k' \} \cap \left( \mathbf{A}^\times(\epsilon, \delta, \eta) \right)^c \right)$$

$$\geq \mathbb{P} \left( v^\Omega \lambda(\eta) U_2'(\epsilon, \delta, \eta) > (1 - C)u \right) \frac{\nu_{k'}^\Omega + C}{\nu^\Omega} - 2C$$

due to $\mathbb{P}(E \backslash F) \geq \mathbb{P}(E) - \mathbb{P}(F)$ and eq. (G.115)eq. (G.117)eq. (G.119) where

$$U_2'(\epsilon, \delta, \eta) \sim \text{Geom} \left( H(\delta/\eta) p \left( -3(\bar{\rho} + \tilde{\rho} + 3)\epsilon, \delta, \eta \right) \right).$$

Using the similar argument leading to eq. (G.121), we are able to show eq. (G.81), eq. (G.83) and conclude the proof. $\qquad \square$

Recall that $\sigma_i(\eta) = \min\{n \geq 0 : X_n \notin \Omega_i\}$ and that value of constants $q_i, q_{i,j}$ are specified via eq. (G.2)-eq. (G.7). Define

$$\tau_i^{\min}(\eta, \epsilon) \triangleq \min\{n \geq \sigma_i(\eta) : X_n^\eta \in \bigcup_j [m_j - 2\epsilon, m_j + 2\epsilon]\}, \tag{G.123}$$

$$J_i(\eta) = j \iff X_{\sigma_i(\eta)}^\eta \in \Omega_j \; \forall j \in [n_{\min}]. \tag{G.124}$$

The following result is simply a restatement of Proposition G.23 under the new system of notations. Despite the reiteration, we still state it here because this is the version that will be used to prove Lemma G.2, which is the key tool for establishing Theorem 1, as well as many other results in Section H.

**Proposition G.25.** *Given $C > 0$ and $i, j \in [n_{min}]$ such that $i \neq j$, the following claims hold for all $\epsilon > 0$ that are sufficiently small:*

$$\limsup_{\eta \downarrow 0} \sup_{u \in (C, \infty)} \sup_{x \in (m_i - 2\epsilon, m_i + 2\epsilon)} \mathbb{P}_x \left( q_i \lambda_i(\eta) \sigma_i(\eta) > u, X_{\sigma_i(\eta)}^\eta \in \Omega_j \right)$$

$$\leq C + \exp \left( -(1 - C)u \right) \frac{q_{i,j} + C}{q_i},$$

$$\liminf_{\eta \downarrow 0} \inf_{u \in (C, \infty)} \inf_{x \in (m_i - 2\epsilon, m_i + 2\epsilon)} \mathbb{P}_x \left( q_i \lambda_i(\eta) \sigma_i(\eta) > u, X_{\sigma_i(\eta)}^\eta \in \Omega_j \right)$$

$$\geq -C + \exp \left( -(1 + C)u \right) \frac{q_{i,j} - C}{q_i},$$

$$\limsup_{\eta \downarrow 0} \sup_{u \in (C, \infty)} \sup_{x \in (m_i - 2\epsilon, m_i + 2\epsilon)} \mathbb{P}_x \left( q_i \lambda_i(\eta) \tau_i^{min}(\eta, \epsilon) > u, X_{\tau_i^{min}(\eta, \epsilon)}^\eta \in \Omega_j \right)$$

$$\leq C + \exp \left( -(1 - C)u \right) \frac{q_{i,j} + C}{q_i},$$

$$\liminf_{\eta \downarrow 0} \inf_{u \in (C, \infty)} \inf_{x \in (m_i - 2\epsilon, m_i + 2\epsilon)} \mathbb{P}_x \left( q_i \lambda_i(\eta) \tau_i^{min}(\eta, \epsilon) > u, X_{\tau_i^{min}(\eta, \epsilon)}^\eta \in \Omega_j \right)$$

$$\geq -C + \exp \left( -(1 + C)u \right) \frac{q_{i,j} - C}{q_i},$$

$$\liminf_{\eta \downarrow 0} \inf_{x \in (m_i - 2\epsilon, m_i + 2\epsilon)} \mathbb{P}_x \left( q_i \lambda_i(\eta) \left( \tau_i^{min}(\eta, \epsilon) - \sigma_i(\eta) \right) < C, \right.$$

$$\left. X_n^\eta \in \Omega_{J_i(\eta)} \; \forall n \in [\sigma_i(\eta), \tau_i^{min}(\eta, \epsilon)] \right) \geq 1 - C.$$

Concluding this section, we apply Proposition G.25 and prove Lemma G.2.

*Proof of Lemma G.2.* Fix some $C \in (0, 1)$, $u > 0$, and some $k, l \in [n_{\min}]$ with $k \neq l$. Let $q_i, q_{i,j}$ be the constants defined in eq. (G.7).

Fix some $C_0 \in \left(0, \frac{C}{n_{\min}} \wedge \frac{q_k}{n_{\min}} C\right)$. Using Proposition G.25, we know that for all $\epsilon$ sufficiently small, we have

$$\limsup_{\eta \downarrow 0} \sup_{x \in (m_k - 2\epsilon, m_k + 2\epsilon)} \mathbb{P}_x \left( q_k \lambda_k(\eta) \sigma_k(\eta) > u, \ X^\eta_{\sigma_k(\eta)} \in \Omega_j \right)$$

$$\leq C_0 + \exp\left( -(1 - C)u \right) \frac{q_{k,j} + C_0}{q_k} \ \forall j \in [n_{\min}].$$

Summing up the inequality above over all $j \in [n_{\min}]$, we can obtain eq. (G.16). The lower bound eq. (G.17) can be established using an identical approach.

In order to show eq. (G.19), note that we can find $C_1 \in (0, u)$ sufficiently small so that

$$-C_1 + \exp\left( -(1 + C_1) \cdot 2C_1 \right) \frac{q_{k,l} - C_1}{q_k} \geq \frac{q_{k,l} - C}{q_k}.$$

Fix such $C_1$. From Proposition G.25, we also know that for all $\epsilon$ small enough, we have

$$\liminf_{\eta \downarrow 0} \inf_{x \in (m_k - 2\epsilon, m_k + 2\epsilon)} \mathbb{P}_x \left( q_k \lambda_k(\eta) \sigma_k(\eta) > u, X^\eta_{\sigma_k(\eta)} \in \Omega_l \right)$$

$$\geq -C_1 + \exp\left( -(1 + C_1) \cdot 2C_1 \right) \frac{q_{k,l} - C_1}{q_k}.$$

Then using $\mathbb{P}_x \left( X^\eta_{\sigma_k(\eta)} \in \Omega_l \right) \geq \mathbb{P}\left( q_k \lambda_k(\eta) \sigma_k(\eta) > u, X^\eta_{\sigma_k(\eta)} \in \Omega_l \right)$ we conclude the proof for eq. (G.19).

Moving on, we show eq. (G.18) in the following way. Note that we can find $C_2 \in (0, u)$ small enough so that

$$2C_2 + \frac{q_{k,l} + C_2}{q_k} < \frac{q_{k,l} + C}{q_k}. \tag{G.125}$$

Fix such $C_2$. Since eq. (G.17) has been established already, we can find some $u_2 > 0$ such that for all $\epsilon$ small enough,

$$\limsup_{\eta \downarrow 0} \sup_{x \in (m_k - 2\epsilon, m_k + 2\epsilon)} \mathbb{P}_x \left( q_k \lambda_k(\eta) \sigma_k(\eta) \leq u_2 \right) < C_2 \tag{G.126}$$

Fix such $u_2$. Meanwhile, fix some $C_3 \in (0, C_2 \wedge u_2)$. From Proposition G.25 we know that for all $\epsilon$ sufficiently small,

$$\limsup_{\eta \downarrow 0} \sup_{x \in (m_k - 2\epsilon, m_k + 2\epsilon)} \mathbb{P}_x \left( q_k \lambda_k(\eta) \sigma_k(\eta) > u_2, \ X^\eta_{\sigma_k(\eta)} \in \Omega_l \right) \tag{G.127}$$

$$\leq C_3 + \exp\left( -(1 - C_3)u_2 \right) \frac{q_{k,l} + C_3}{q_k}$$

$$\leq C_2 + \frac{q_{k,l} + C_2}{q_k}. \tag{G.128}$$

Lastly, observe the following decomposition of events (for any $x \in \Omega_k$)

$$\mathbb{P}_x \left( X^\eta_{\sigma_k(\eta)} \in \Omega_l \right) \leq \mathbb{P}_x \left( q_k \lambda_k(\eta) \sigma_k(\eta) \leq u_2 \right) + \mathbb{P}_x \left( q_k \lambda_k(\eta) \sigma_k(\eta) > u_2, \ X^\eta_{\sigma_k(\eta)} \in \Omega_l \right).$$

Combining this bound with eq. (G.125)-eq. (G.128), we complete the proof. $\qquad\square$

## H    PROOFS FOR SECTION 2.2

In this section, we show that gradient clipping scheme effectively partitions the entire optimization landscape of $f$ into different regions based on the radius $r_i$ and minimum jump number $l_i^*$ for each attraction field $\Omega_i$. Furthermore, when staying in each region, the behavior of SGD iterates closely

resembles a Markov chain that *only* visits *wider* attraction fields in this region. We exclude the trivial case where $n_{\min} = 1$ and there is only one attraction field.

This structure is as follows. First we present some key lemmas that can be used to prove the Theorem 2-3 in the main paper. Then we devote the rest of the section to establish those lemmas. In order to prove Theorem 2, we will make use of the following lemma, where we show that the type of claim in Theorem 2 is indeed valid if we look at a much shorter time interval. Then when we move onto the proof of Theorem 2, it suffices to partition the entire horizon into pieces of these short time intervals, on each of which we analyze the dynamics of SGD respectively.

**Lemma H.1.** *Assume the graph $\mathcal{G}$ is **irreducible**, and let $\epsilon > 0, \delta > 0$ be any positive real numbers. For the following random variables (indexed by $\eta$)*

$$V^{small}(\eta, \epsilon, t) \triangleq \frac{1}{\lfloor t/\lambda^{large}(\eta) \rfloor} \int_0^{\lfloor t/\lambda^{large}(\eta) \rfloor} \mathbb{1}\Big\{ X_{\lfloor u \rfloor}^\eta \in \bigcup_{j: m_j \notin M^{large}} \Omega_j \Big\} du, \qquad \text{(H.1)}$$

*the following claim holds for any sufficiently small $t > 0$:*

$$\limsup_{\eta \downarrow 0} \sup_{x \in [-L, L]} \mathbb{P}_x\Big( V^{small}(\eta, \epsilon, t) > \epsilon \Big) \le 5\delta$$

*Proof of Theorem 2.* It suffices to show that for any $t > 0, \kappa > 1 + (\alpha - 1)l^{large}, \epsilon > 0, \delta \in (0, \epsilon)$, we have

$$\limsup_{\eta \downarrow 0} \mathbb{P}_x\Big( V^*(\eta, t, \kappa) > 3\epsilon \Big) < \delta$$

for

$$V^*(\eta, t, \kappa) \triangleq \frac{1}{\lfloor t/\eta^\kappa \rfloor} \int_0^{\lfloor t/\eta^\kappa \rfloor} \mathbb{1}\Big\{ X_{\lfloor u \rfloor}^\eta(x) \in \bigcup_{j: m_j \notin M^{large}} \Omega_j \Big\} du.$$

Let us fix some $\epsilon > 0, \delta \in (0, \epsilon)$. First, let

$$N(\eta) = \Big\lceil \frac{\lfloor t/\eta^\kappa \rfloor}{\lfloor t/\lambda^{large}(\eta) \rfloor} \Big\rceil.$$

The regularly varying nature of $H$ implies that $\lambda^{large}(\eta) \in \mathcal{RV}_{1+l^{large}(\alpha-1)}(\eta)$. Since $\kappa > 1 + l^{large}(\alpha - 1)$, we know that $\lim_{\eta \downarrow 0} N(\eta) = \infty$. Next, due to Lemma H.1, we can find $t_0 > 0$ and $\bar\eta > 0$ such that for any $\eta \in (0, \bar\eta)$

$$\sup_{y \in [-L, L]} \mathbb{P}_y(V^{small}(\eta, \epsilon, t_0) > \epsilon) < \delta. \qquad \text{(H.2)}$$

For any $k \ge 1$, define

$$V_k(\eta) \triangleq \frac{1}{\lfloor t_0/\lambda^{large}(\eta) \rfloor} \int_{(k-1)\lfloor t_0/\lambda^{large}(\eta) \rfloor}^{k\lfloor t_0/\lambda^{large}(\eta) \rfloor} \mathbb{1}\Big\{ X_{\lfloor u \rfloor}^\eta \in \bigcup_{j: m_j \notin M^{large}} \Omega_j \Big\} du. \qquad \text{(H.3)}$$

It is clear from its definition that $V_k$ stands for the proportion of time that the SGD iterates are outside of *large* attraction fields on the interval $[(k-1)\lfloor \frac{t_0}{\lambda^{large}(\eta)} \rfloor, k\lfloor \frac{t_0}{\lambda^{large}(\eta)} \rfloor]$. From eq. (H.2) and Markov property, one can see that for any $\eta \in (0, \bar\eta)$

$$\sup_{x \in [-L, L]} \mathbb{P}_x(V_k(\eta) > \epsilon \mid X_0^\eta, \cdots, X_{(k-1)\lfloor t_0/\lambda^{large}(\eta) \rfloor}^\eta) \le \delta$$

uniformly for all $k \ge 1$. Now define $K(\eta) \triangleq \#\{n = 1, 2, \cdots, N(\eta) : V_k(\eta) > \epsilon\}$. By a simple stochastic dominance argument, we have

$$\sup_{x \in [-L, L]} \mathbb{P}_x(K(\eta) \ge j) \le \mathbb{P}(\text{Binomial}(N(\eta), \delta) \ge j) \quad \forall j = 1, 2, \cdots.$$

Meanwhile, strong law of large numbers implies the existence of some $\bar{\eta}_1 > 0$ such that $\mathbb{P}\left(\frac{\text{Binomial}(N(\eta),\delta)}{N(\eta)} > 2\delta\right) < \delta$ for all $\eta \in (0, \bar{\eta}_1)$, thus

$$\sup_{|x| \leq L} \mathbb{P}_x(K(\eta)/N(\eta) > 2\delta) \leq \delta \ \ \forall \eta \in (0, \bar{\eta}_1 \wedge \bar{\eta}).$$

Lastly, from the definition of $K(\eta)$ and $N(\eta)$, we know that for all the $N(\eta)$ intervals $[(k-1)\lfloor \frac{t_0}{\lambda^{\text{large}}(\eta)} \rfloor, k \lfloor \frac{t_0}{\lambda^{\text{large}}(\eta)} \rfloor]$ with $k \in [N(\eta)]$, only on $K(\eta)$ of them did the SGD iterates spent more then $\epsilon$ proportion of time outside of the *large* attraction fields, hence

$$V^*(\eta, t, \kappa) \leq \epsilon + \frac{K(\eta)}{N(\eta)}.$$

In summary, we now have

$$\mathbb{P}_x(V^*(\eta, t, \kappa) > 3\epsilon) < \delta$$

for all $\eta \in (0, \bar{\eta}_1 \wedge \bar{\eta})$. This concludes the proof. $\qquad \square$

When introducing Theorem 3 in the main paper, we stated that the results of eliminating sharp minima can be extended to the more general reducible case. Here we present the corresponding theoretical results in Theorem H.2 and H.3 below. The main message can be summarized as follows: (a) SGD with truncated and heavy-tailed noise naturally partitions the entire training landscape into different regions; (b) In each region, the dynamics of $X_n^\eta$ for small $\eta$ closely resemble that of a continuous-time Markov chain that *only* visits local minima; (3) In particular, any sharp minima within each region is almost completely avoided by SGD.

When the typical transition graph (see Definition 1 in the main paper) is not irreducible, there will be multiple communication classes on the graph. Suppose that there are $K$ communication classes $G_1, \cdots, G_K$. From now on, we zoom in on a specific communication class $G \in \{G_1, \cdots, G_K\}$. For this communication class $G$, define $l_G^* \triangleq \max\{l_i^* : i = 1, 2, \cdots, n_{\min}; m_i \in G\}$. For each local minimum $m_i \in G$, we call its attraction field $\Omega_i$ a *large* attraction field if $l_i^* = l_G^*$, and a *small* attraction field if $l_i^* < l_G^*$. We have thus classified all $m_i$ in $G$ into two groups: the ones in large attraction fields $m_1^{\text{large}}, \cdots, m_{i_G}^{\text{large}}$ and the ones in small attraction fields $m_1^{\text{small}}, \cdots, m_{i_G'}^{\text{small}}$. Also, define a scaling function $\lambda_G$ associated with $G$ as $\lambda_G(\eta) \triangleq H(1/\eta)\left(\frac{H(1/\eta)}{\eta}\right)^{l_G^*-1}$.

**Theorem H.2.** *Under Assumptions 1-3, if $G$ is **absorbing**, then there exists a continuous-time Markov chain $Y$ on $\{m_1^{\text{large}}, \cdots, m_{i_G}^{\text{large}}\}$ such that for any $x \in \Omega_i, |x| \leq L$ (where $i \in \{1, 2, \cdots, n_{\min}\}$) with $m_i \in G$, and*

$$X_{\lfloor t/\lambda_G(\eta) \rfloor}^\eta(x) \to Y_t(\pi_G(m_i)) \quad \text{as } \eta \downarrow 0$$

*in the sense of finite-dimensional distributions, where $\pi_G$ is a random mapping satisfying (1) $\pi_G(m) \equiv m$ if $m \in \{m_1^{\text{large}}, \cdots, m_{i_G}^{\text{large}}\}$; (2) $\pi_G(m)$ is a random variable that only takes value in $\{m_1^{\text{large}}, \cdots, m_{i_G}^{\text{large}}\}$ if $m \in \{m_1^{\text{small}}, \cdots, m_{i_G'}^{\text{small}}\}$.*

We stress that Theorem 3 in the main paper follows immediately from Theorem H.2 above.

Next, to state the corresponding result for a **transient** communication class $G$, we introduce a couple of extra definitions. We consider a version of $X_n^\eta$ that is killed when $X_n^\eta$ leaves $G$. Define stopping time

$$\tau_G(\eta) \triangleq \min\{n \geq 0 : X_n^\eta \notin \bigcup_{i:m_i \in G} \Omega_i\} \qquad (\text{H.4})$$

as the first time the SGD iterates leave all attraction fields in $G$, and we use a cemetery state † to construct the following process $X_n^{\dagger,\eta}$ as a version of $X_n^\eta$ with killing at $\tau_G$:

$$X_n^{\dagger,\eta} = \begin{cases} X_n^\eta & \text{if } n < \tau_G(\eta), \\ \dagger & \text{if } n \geq \tau_G(\eta). \end{cases} \qquad (\text{H.5})$$

**Theorem H.3.** *Under Assumptions 1-3, if $G$ is **transient**, then there exists a continuous-time Markov chain $Y$ **with killing** that has state space $\{m_1^{large}, \cdots, m_{i_G}^{large}, \dagger\}$ (we say the Markov chain $Y$ is killed when it enters the absorbing cemetery state $\dagger$) such that for any $x \in \Omega_i, |x| \le L$ (where $i \in \{1, 2, \cdots, n_{min}\}$) with $m_i \in G$, and*

$$X^{\dagger, \eta}_{\lfloor t/\lambda_G(\eta) \rfloor}(x) \to Y_t(\pi_G(m_i)) \quad as \ \eta \downarrow 0$$

*in the sense of finite-dimensional distributions, where $\pi_G$ is a random mapping satisfying (1) $\pi_G(m) \equiv m$ if $m \in \{m_1^{large}, \cdots, m_{i_G}^{large}\}$; (2) $\pi_G(m)$ is a random variable that only takes value in $\{m_1^{large}, \cdots, m_{i_G}^{large}, \dagger\}$ if $m \in \{m_1^{small}, \cdots, m_{i'_G}^{small}\}$.*

To show Theorem H.2 and H.3, we introduce the following concepts. First, we consider the case where the SGD iterates $X_n^\eta$ is initialized on the communication class $G$ and $G$ is absorbing. For some $\Delta > 0, \eta > 0$, define (let $B(u, v) \triangleq [u - v, u + v]$)

$$\sigma_0^G(\eta, \Delta) \triangleq \min\{n \ge 0 : X_n^\eta \in \bigcup_{i: \, m_i \in G} B(m_i, 2\Delta)\} \tag{H.6}$$

$$\tau_0^G(\eta, \Delta) \triangleq \min\{n \ge \sigma_0^G(\eta, \Delta) : X_n^\eta \in \bigcup_{i: m_i \notin G^{\text{small}}} B(m_i, 2\Delta)\} \tag{H.7}$$

$$I_0^G(\eta, \Delta) = j \iff X_{\tau_0^G}^\eta \in B(m_j, 2\Delta), \quad \widetilde{I}_0^G(\eta, \Delta) = j \iff X_{\sigma_0^G}^\eta \in B(m_j, 2\Delta) \tag{H.8}$$

$$\sigma_k^G(\eta, \Delta) \triangleq \min\{n > \tau_{k-1}^G(\eta, \Delta) : X_n^\eta \in \bigcup_{i: m_i \in G, \, i \ne I_{k-1}^G} B(m_i, 2\Delta)\} \, \forall k \ge 1 \tag{H.9}$$

$$\tau_k^G(\eta, \Delta) \triangleq \min\{n \ge \sigma_{k-1}^G(\eta, \Delta) : X_n^\eta \in \bigcup_{i: m_i \notin G^{\text{small}}} B(m_i, 2\Delta)\} \, \forall k \ge 1 \tag{H.10}$$

$$I_k^G(\eta, \Delta) = j \iff X_{\tau_k^G}^\eta \in B(m_j, 2\Delta), \quad \widetilde{I}_k^G(\eta, \Delta) = j \iff X_{\sigma_k^G}^\eta \in B(m_j, 2\Delta) \forall k \ge 1. \tag{H.11}$$

Intuitively speaking, at each $\tau_k^G$ the SGD iterates visits a minimizer that is not in a *small* attraction field on $G$, and we use $I_k^G$ to mark the label of that large attraction field. Stopping time $\sigma_k^G$ is the first time that SGD visits a minimizer that is *different* from the one visited at $\tau_k^G$, and $\tau_{k+1}^G$ is the first time that a minimizer not in a small attraction field of $G$ is visited again since $\sigma_k^G$ (and including $\sigma_k^G$). It is worth mentioning that, under this definition, we could have $I_k^G = I_{k+1}^G$ for any $k \ge 0$. Meanwhile, define the following process that only keeps track of the updates on the labels $(I_k^G)_{k \ge 0}$ instead of the information of the entire trajectory of $(X_n^\eta)_{n \ge 0}$:

$$\hat{X}_n^{\eta, \Delta} = \begin{cases} m_{I_k^G} & \text{if } \exists k \ge 0 \text{ such that } \tau_k^G \le n < \tau_{k+1}^G \\ 0 & \text{otherwise} \end{cases} \tag{H.12}$$

In other words, when $n < \tau_0^G$ we simply let $\hat{X}_n^{\eta, \Delta} = 0$, otherwise it is equal to the latest "marker" for the last visited wide minimum up until step $n$. This marker process $\hat{X}$ jumps between the different minimizers of the large attractions in $G$. In particular, if for some $n$ we have $X_n^\eta \in B(m_j, 2\Delta)$ for some $j$ with $m_j \in G^{\text{large}}$, then we must have $\hat{X}_n^{\eta, \Delta} = m_j$, which implies that, in this case, $\hat{X}_n^{\eta, \Delta}$ indeed indicates the location of $X_n^\eta$.

Note that results in Theorem H.2 and H.3 concern a *scaled* version of $X^\eta$. Here we also define the corresponding *scaled* version of the processes

$$X_t^{*, \eta} \triangleq X_{\lfloor t/\lambda_G(\eta) \rfloor}^\eta \tag{H.13}$$

$$\hat{X}_t^{*, \eta, \Delta} \triangleq \hat{X}_{\lfloor t/\lambda_G(\eta) \rfloor}^{\eta, \Delta}, \tag{H.14}$$

a mapping $\mathbf{T}^*(n, \eta) \triangleq n\lambda_G(\eta)$ that translates a step $n$ to the corresponding timestamp for the scaled processes, and the following series of scaled stopping times

$$\tau_k^*(\eta, \Delta) = \mathbf{T}^*\left(\tau_k^G(\eta, \Delta), \eta\right), \quad \sigma_k^*(\eta, \Delta) = \mathbf{T}^*\left(\sigma_k^G(\eta, \Delta), \eta\right). \tag{H.15}$$

Before presenting the proof of Theorem H.2 and H.3, we make several preparations. First, our proof is inspired by ideas in Pavlyukevich (2005) and here we provide a briefing. At any time $t > 0$, if we can show that $X_t^{*,\eta}$ is almost always in set $\bigcup_{i:m_i \in G^{\text{large}}} B(m_i, 2\Delta)$ (so the SGD iterates is almost always close to a minimizer in a large attraction field), then the marker process $\hat{X}_t^{*,\eta,\Delta}$ is a pretty accurate indicator of the location of $X_t^{*,\eta}$, so it suffices to show that the marker process $\hat{X}_t^{*,\eta,\Delta}$ converges to a continuous-time Markov chain $Y$.

Second, we construct the limiting process $Y$ and the random mapping $\pi_G$ before utilizing them in Theorem H.2 and H.3. As an important building block for this purpose, we start by considering the following discrete time Markov chain (DTMC) on the entire graph $\mathcal{G} = (V, E)$. Let $\mathbf{P}^{DTMC}$ be a transition matrix with $\mathbf{P}^{DTMC}(m_i, m_j) = \mu_i(E_{i,j})/\mu_i(E_i)$ for all $j \neq i$, and $Y^{DTMC} = (Y_j^{DTMC})_{j \geq 0}$ be the DTMC induced by the said transition matrix. Let

$$T_G^{DTMC} \triangleq \min\{j \geq 0 : Y_j^{DTMC} \notin G^{\text{small}}\} \tag{H.16}$$

be the first time this DTMC visits a large attraction field on the communication class $G$, or escapes from $G$. Lastly, define (for any $j$ such that $m_j \notin G^{\text{small}}$)

$$p_{i,j} \triangleq \mathbb{P}\big(Y_{T_G^{DTMC}}^{DTMC}(m_i) = m_j\big) \tag{H.17}$$

as the probability that the first large attraction field on $G$ visited by $Y^{DTMC}$ is $m_j$ when initialized at $m_i$.

We add a comment regarding the stopping times $T_G^{DTMC}$ and probabilities $p_{i,j}$ defined above. In the case that $G$ is absorbing, we have $Y_j^{DTMC}(m_i) \in G$ for all $j \geq 0$ if $m_i \in G$. Therefore, in this case, given any $i$ with $m_i \in G$, we must have

$$T_G^{DTMC} = \min\{j \geq 0 : Y_j^{DTMC}(m_i) \in G^{\text{large}}\}, \qquad \sum_{j:\, m_j \in G^{\text{large}}} p_{i,j} = 1.$$

On the contrary, when $G$ is transient we may have $\sum_{j:\, m_j \in G^{\text{large}}} p_{i,j} < 1$ and $\sum_{j:\, m_j \notin G} p_{i,j} > 0$. Lastly, whether $G$ is absorbing or transient, we always have $p_{i,j} = \mathbb{1}\{i = j\}$ if $m_i \in G^{\text{large}}$.

Next, consider the following definition of (continuous-time) jump processes.

**Definition H.1.** *A continuous-time process $Y_t$ on $\mathbb{R}$ is a $\big((U_j)_{j \geq 0}, (V_j)_{j \geq 0}\big)$ **jump process** if*

$$Y_t = \begin{cases} 0 & \text{if } t < U_0 \\ \sum_{j \geq 0} V_j \mathbb{1}_{[U_0 + U_1 + \cdots + U_j,\ U_0 + U_1 + \cdots + U_{j+1})}(t) & \text{otherwise} \end{cases},$$

*where $(U_j)_{j \geq 0}$ is a sequence of non-negative random variables such that $U_j > 0\ \forall j \geq 1$ almost surely, and $(V_j)_{j \geq 0}$ is a sequence of random variables in $\mathbb{R}$.*

Obviously, the definition above implies that $Y_t = V_j$ for any $t \in [U_j, U_{j+1})$.

Now we are ready to construct the limiting continuous-time Markov chain $Y$. To begin with, we address the case where $G$ is absorbing. For any $m' \in G^{\text{large}}$, let $Y(m')$ be a $\big((S_k)_{k \geq 0}, (W_k)_{k \geq 0}\big)$-jump process where $S_0 = 0, W_0 = m'$ and (for all $k \geq 0$ and $i, j$ with $m_i \in G^{\text{large}}, m_j \notin G^{\text{small}}$)

$$\mathbb{P}\Big(W_{k+1} = m_j,\ S_{k+1} > t \,\Big|\, W_k = m_i,\ (W_l)_{l=0}^{k-1},\ (S_l)_{l=0}^{k}\Big) \tag{H.18}$$

$$= \mathbb{P}\Big(W_{k+1} = m_j,\ S_{k+1} > t \,\Big|\, W_k = m_i\Big) = \exp(-q_i t)\frac{q_{i,j}}{q_i} \quad \forall t > 0 \tag{H.19}$$

where

$$q_i = \mu_i(E_i) \tag{H.20}$$

$$q_{i,j} = \mathbb{1}\{i \neq j\}\mu_i(E_{i,j}) + \sum_{k:\, m_k \in G^{\text{small}}} \mu_i(E_{i,k})p_{k,j} \tag{H.21}$$

and $p_{k,j}$ is defined in eq. (H.17). In other words, conditioning on $W_k = m_i$, the time until next jump $S_{k+1}$ and the jump location $W_{k+1}$ are independent, where $S_{k+1}$ is $\text{Exp}(q_i)$ and $W_{k+1} = m_j$ with

probability $q_{i,j}/q_i$. First, it is easy to see that $Y$ is a continuous-time Markov chain. Second, under this definition $Y$ is allowed to have some *dummy* jumps where $W_k = W_{k+1}$: in this case the process $Y_t$ does not move to a different minimizer after the $k+1$-th jump, and by inspecting the path of $Y$ we cannot tell that this dummy jump has occurred. As a result, that generator $Q$ of this Markov chain admits the form (for all $i \neq j$ with $m_i, m_j \in G^{\text{large}}$)

$$Q_{i,i} = - \sum_{k: k \neq i, \ m_k \in G^{\text{large}}} q_{i,k}, \ Q_{i,j} = q_{i,j}.$$

Moreover, define the following random function $\pi_G(\cdot)$ such that for any $m_i \in G$,

$$\pi_G(m_i) = \begin{cases} m_j & \text{with probability } q_{i,j}/q_i \text{ if } m_i \in G^{\text{small}} \\ m_i & \text{if } m_i \in G^{\text{large}} \end{cases} \tag{H.22}$$

By $Y(\pi_G(m_i))$ we refer to the version of the Markov chain $Y$ where we randomly initialize $W_0 = \pi_G(m_i)$. The following lemma is the key tool for proving Theorem H.2.

**Lemma H.4.** *Assume that the communication class $G$ is absorbing. Given any $m_i \in G$, $x \in \Omega_i$, finitely many real numbers $(t_l)_{l=1}^{k'}$ such that $0 < t_1 < t_2 < \cdots < t_{k'}$, and a sequence of strictly positive real numbers $(\eta_n)_{n \geq 1}$ with $\lim_{n \to 0} \eta_n = 0$, there exists a sequence of strictly positive real numbers $(\Delta_n)_{n \geq 1}$ with $\lim_n \Delta_n = 0$ such that*

- *As $n$ tends to $\infty$,*
$$\left( \hat{X}_{t_1}^{*,\eta_n,\Delta_n}(x), \cdots, \hat{X}_{t_{k'}}^{*,\eta_n,\Delta_n}(x) \right) \Rightarrow \left( Y_{t_1}(\pi_G(m_i)), \cdots, Y_{t_{k'}}(\pi_G(m_i)) \right) \tag{H.23}$$

- *For all $k \in [k']$,*
$$\lim_{n \to \infty} \mathbb{P}_x \left( X_{t_k}^{*,\eta_n} \notin \bigcup_{j: \ m_j \in G^{large}} B(m_j, \Delta_n) \right) = 0. \tag{H.24}$$

Now we address the case where $G$ is transient, let $\dagger$ be a real number such that $\dagger \notin [-L, L]$, and we use $\dagger$ as the cemetery state since the processes $X_n^{\eta}$ or $X_t^{*,\eta}$ are restricted on $[-L, L]$. Recall the definition of $\tau_G$ defined in eq. (H.4). Analogous to the process $X^{\dagger}$ in eq. (H.5), we can also define

$$X_t^{\dagger,*,\eta} = \begin{cases} X_t^{*,\eta} & \text{if } t < \mathbf{T}^*(\tau_G(\eta), \eta) \\ \dagger & \text{otherwise} \end{cases}, \ \hat{X}_t^{\dagger,*,\eta,\Delta} = \begin{cases} \hat{X}_t^{*,\eta,\Delta} & \text{if } t < \mathbf{T}^*(\tau_G(\eta), \eta) \\ \dagger & \text{otherwise,} \end{cases} \tag{H.25}$$

Next, analogous to $\tau_G$, consider the stopping time

$$\tau_G^Y \triangleq \min\{t > 0 : Y_t \notin G\}.$$

When $G$ is transient, due to the construction of $Y$ we know that $\tau_G^Y < \infty$ almost surely. The introduction of $\tau_G^Y$ allows us to define

$$Y_t^{\dagger} = \begin{cases} Y_t & \text{if } t < \tau_G^Y \\ \dagger & \text{otherwise.} \end{cases} \tag{H.26}$$

The following Lemma will be used to prove Theorem H.3.

**Lemma H.5.** *Assume that the communication class $G$ is transient. Given any $m_i \in G$, $x \in \Omega_i$, finitely many real numbers $(t_l)_{l=1}^{k'}$ such that $0 < t_1 < t_2 < \cdots < t_{k'}$, and a sequence of strictly positive real numbers $(\eta_n)_{n \geq 1}$ with $\lim_{n \to 0} \eta_n = 0$, there exists a sequence of strictly positive real numbers $(\Delta_n)_{n \geq 1}$ with $\lim_n \Delta_n = 0$ such that*

- *As $n$ tends to $\infty$,*
$$\left( \hat{X}_{t_1}^{\dagger,*,\eta_n,\Delta_n}(x), \cdots, \hat{X}_{t_{k'}}^{\dagger,*,\eta_n,\Delta_n}(x) \right) \Rightarrow \left( Y_{t_1}^{\dagger}(\pi_G(m_i)), \cdots, Y_{t_{k'}}^{\dagger}(\pi_G(m_i)) \right) \tag{H.27}$$

- *For all $k \in [k']$,*
$$\lim_{n \to \infty} \mathbb{P}_x \left( X_{t_k}^{\dagger,*,\eta_n} \notin \bigcup_{j: \ m_j \in G^{large}} B(m_j, \Delta_n) \text{ and } X_{t_k}^{\dagger,*,\eta_n} \neq \dagger \right) = 0. \tag{H.28}$$

*Proof of Theorem H.2 and H.3.* We first address the case where $G$ is absorbing. Arbitrarily choose some $\Delta > 0$, a sequence of strictly positive real numbers $(\eta_n)_{n \geq 1}$ with $\lim_n \eta_n = 0$, a positive integer $k'$, a series of real numbers $(t_j)_{j=1}^{k'}$ with $0 < t_1 < \cdots < t_{k'}$, and a sequence $(w_j)_{j=1}^{k'}$ with $w_j \in G^{\text{large}}$ for all $j \in [k']$. It suffices to show that

$$\lim_n \mathbb{P}_x\Big(X_{t_k}^{*,\eta_n} \in B(w_k, \Delta) \ \forall k \in [k']\Big) = \mathbb{P}\Big(Y_{t_k}(\pi_G(m_i)) = w_k \ \forall k \in [k']\Big).$$

Using Lemma H.4, we can find a sequence of strictly positive real numbers $(\Delta_n)_{n \geq 1}$ with $\lim_n \Delta_n = 0$ such that eq. (H.23) and eq. (H.24) hold. From the weak convergence in eq. (H.23), we only need to show

$$\lim_n \mathbb{P}_x\Big(X_{t_k}^{*,\eta_n} \notin B(\hat{X}_{t_k}^{*,\eta_n,\Delta_n}, \Delta)\Big) = 0 \ \forall k \in [k'].$$

For all $n$ large enough, we have $2\Delta_n < \Delta$. For such large $n$, observe that

$$\mathbb{P}_x\Big(X_{t_k}^{*,\eta_n} \notin B(\hat{X}_{t_k}^{*,\eta_n,\Delta_n}, \Delta)\Big)$$

$$\leq \mathbb{P}_x\Big(X_{t_k}^{*,\eta_n} \notin \bigcup_{j: m_j \in G^{\text{large}}} B(m_j, 2\Delta_n)\Big)$$

(due to definition of the marker process $\hat{X}$, see eq. (H.6)-eq. (H.11) and eq. (H.12)-eq. (H.14))

$$\leq \mathbb{P}_x\Big(X_{t_k}^{*,\eta_n} \notin \bigcup_{j: m_j \in G^{\text{large}}} B(m_j, \Delta_n)\Big)$$

and by applying eq. (H.24) we conclude the proof for Theorem H.2.

The proof of Theorem H.3 is almost identical, with the only modification being that we apply Lemma H.5 instead of Lemma H.4. In doing so, we are able to find a sequence of $(\Delta_n)_{n \geq 1}$ with $\lim_n \Delta_n = 0$ such that eq. (H.27) and eq. (H.28) hold. Given the weak convergence claim in eq. (H.27), it suffices to show that

$$\lim_n \mathbb{P}_x\Big(X_{t_k}^{\dagger*,\eta_n} \notin B(\hat{X}_{t_k}^{\dagger,*,\eta_n,\Delta_n}, \Delta)\Big) = 0 \ \forall k \in [k'].$$

If $X_{t_k}^{\dagger,*,\eta_n} = \dagger$, we must have $\hat{X}_{t_k}^{\dagger,*,\eta_n,\Delta_n} = \dagger$ as well. Therefore, for all $n$ large enough so that $3\Delta_n < \Delta$,

$$\mathbb{P}_x\Big(X_{t_k}^{\dagger*,\eta_n} \notin B(\hat{X}_{t_k}^{\dagger,*,\eta_n,\Delta_n}, \Delta)\Big)$$

$$= \mathbb{P}_x\Big(X_{t_k}^{\dagger*,\eta_n} \notin B(\hat{X}_{t_k}^{\dagger,*,\eta_n,\Delta_n}, \Delta), \ X_{t_k}^{\dagger*,\eta_n} \neq \dagger\Big)$$

$$\leq \mathbb{P}_x\Big(X_{t_k}^{\dagger*,\eta_n} \notin \bigcup_{j: m_j \in G^{\text{large}}} B(m_j, 2\Delta_n), \ X_{t_k}^{\dagger*,\eta_n} \neq \dagger\Big)$$

$$\leq \mathbb{P}_x\Big(X_{t_k}^{\dagger*,\eta_n} \notin \bigcup_{j: m_j \in G^{\text{large}}} B(m_j, \Delta_n), \ X_{t_k}^{\dagger*,\eta_n} \neq \dagger\Big).$$

Apply eq. (H.28) and we conclude the proof. $\qquad \square$

## H.1    PROOF OF LEMMA H.1

First, we introduce another dichotomy for *small* and *large* noises. For any $\widetilde{\gamma} > 0$ and any learning rate $\eta > 0$, we say that a noise $Z_n$ is small if

$$\eta|Z_n| > \eta^{\widetilde{\gamma}}$$

and we say $Z_n$ is large otherwise. For this new classification of small and large noises, we introduce the following notations and definitions:

$$Z_n^{\leq,\widetilde{\gamma},\eta} = Z_n \mathbb{1}\{\eta|Z_n| \leq \eta^{\widetilde{\gamma}}\}, \tag{H.29}$$

$$Z_n^{>,\widetilde{\gamma},\eta} = Z_n \mathbb{1}\{\eta|Z_n| > \eta^{\widetilde{\gamma}}\}, \tag{H.30}$$

$$\widetilde{T}_1^{\eta}(\widetilde{\gamma}) \triangleq \min\{n \geq 1 : \eta|Z_n| > \eta^{\widetilde{\gamma}}\}. \tag{H.31}$$

Similar to Lemma G.12, the following result is a direct application of Lemma G.11, and shows that it is rather unlikely to observe large perturbation that are caused only by *small* noises. Specifically, since $\alpha > 1$ we can always find

$$\widetilde{\gamma} \in (0, (1 - \frac{1}{\alpha} \wedge \frac{1}{2})$$
$$\beta \in \big(1, (2 - 2\widetilde{\gamma}) \wedge \alpha(1 - \widetilde{\gamma})\big).$$

Now in Lemma G.11, if we let $\Delta = \widetilde{\gamma}, \widetilde{\Delta} = \Delta/2$ and $\epsilon = \delta = 1$ (in other words, $u(\eta) = 1/\eta^{1-\widetilde{\gamma}}$, $v(\eta) = \eta^{\widetilde{\gamma}/2}$), then for any positive integer $j$ the condition eq. (G.52) is satisfied, allowing us to draw the following conclusion immediately as a corollary from Lemma G.11.

**Lemma H.6.** *Given $N > 0$ and*

$$\widetilde{\gamma} \in (0, (1 - \frac{1}{\alpha}) \wedge \frac{1}{2}), \ \beta \in \big(1, (2 - 2\widetilde{\gamma}) \wedge (\alpha - \alpha\widetilde{\gamma})\big),$$

*we have (as $\eta \downarrow 0$)*

$$\mathbb{P}\Big(\max_{j=1,2,\cdots,\lceil(1/\eta)^{\beta}\rceil} \eta|Z_1^{\leq,\widetilde{\gamma},\eta} + \cdots + Z_j^{\leq,\widetilde{\gamma},\eta}| > \eta^{\widetilde{\gamma}/2}\Big) = o(\eta^N).$$

The flavor of the next lemma is similar to that of Lemma G.13. Specifically, we show that, with high probability, the SGD iterates would quickly return to the local minimum as long as they start from somewhere that are not too close the boundary of an attraction field (namely, the points $s_1, s_2, \cdots, s_{n_{\min}}$). To this end, we consider a refinement of function $\hat{t}(\cdot)$ defined in eq. (G.38). For any $i = 1, 2, \cdots, n_{\min}$, any $x \in \Omega_i$ and any $\eta > 0, \gamma \in (0, 1)$, we can define the return time to $\eta^{\gamma}-$neighborhood for the ODE $\mathbf{x}^{\eta}$ as

$$\hat{t}_{\gamma}^{(i)}(x, \eta) \triangleq \min\{t \geq 0 : |\mathbf{x}^{\eta}(t, x) - m_i| \leq \eta^{\gamma}\}.$$

Given the bound in eq. (G.37) (which is stated for a specific attraction field) and the fact that there only exists finitely many attraction fields, we know the existence of some $c_2 < \infty$ such that for any $i = 1, 2, \cdots, n_{\min}$, any $\eta > 0$, any $\gamma \in (0, 1)$ and any $x \in \Omega_i$ such that $|x - s_i| \vee |x - s_{i-1}| > \eta^{\gamma}$, we have

$$\hat{t}_{\gamma}^{(i)}(x, \eta) \leq c_2 \gamma \log(1/\eta)/\eta$$

and define function $t^{\uparrow}$ as

$$t^{\uparrow}(\eta, \gamma) \triangleq c_2 \gamma \log(1/\eta).$$

Lastly, define the following stopping time for any $i = 1, 2, \cdots, n_{\min}$, any $x \in \Omega_i$ and any $\Delta > 0$

$$T_{\text{return}}^{(i)}(\eta, \Delta) \triangleq \min\{n \geq 0 : X_n^{\eta}(x) \in B(m_i, 2\Delta)\}$$

where we adopt the notation $B(u, v) \triangleq [u - v, u + v]$ for the $v-$neighborhood around point $u$.

**Lemma H.7.** *Given*

$$\widetilde{\gamma} \in (0, (1 - \frac{1}{\alpha}) \wedge (\frac{1}{2})), \ \gamma \in (0, \frac{\widetilde{\gamma}}{16Mc_2} \wedge \frac{\widetilde{\gamma}}{4}),$$

*and any $i = 1, 2, \cdots, n_{min}$, any $\Delta > 0$, we have*

$$\liminf_{\eta\downarrow 0} \inf_{x\in\Omega_i:|x-s_{i-1}|\vee|x-s_i|\geq 2\eta^{\gamma}} \mathbb{P}_x\Big(T_{return}^{(i)}(\eta, \Delta) \leq \frac{2c_2\gamma\log(1/\eta)}{\eta}, \ X_n^{\eta} \in \Omega_i \ \forall n \leq T_{return}^{(i)}(\eta, \Delta)\Big)$$
$$= 1.$$

*Proof.* Throughout this proof, we only consider $\eta$ small enough such that

$$2c_2\gamma\log(1/\eta)/\eta < \lceil(1/\eta)^{\beta}\rceil, \ \eta M \leq \eta^{2\widetilde{\gamma}}, \ 2\eta^{\gamma} < \Delta/2, \ 2\eta^{\widetilde{\gamma}/4} < \eta^{\gamma}. \tag{H.32}$$

The condition above holds for all $\eta > 0$ sufficiently small because $\beta > 1$, $2\widetilde{\gamma} < 1$, and $\gamma < \widetilde{\gamma}/4$. Also, fix some $\beta \in \big(1, (2 - 2\widetilde{\gamma}) \wedge (\alpha - \alpha\widetilde{\gamma})\big)$

Define the following events

$$A_1^\times(\eta) \triangleq \Big\{ \max_{j=1,2,\cdots,\lceil(1/\eta)^\beta\rceil} \eta|Z_1^{\leq,\widetilde{\gamma},\eta} + \cdots + Z_j^{\leq,\widetilde{\gamma},\eta}| > \eta^{\widetilde{\gamma}/2} \Big\}$$

$$A_2^\times(\eta) \triangleq \{\widetilde{T}_1^\eta(\widetilde{\gamma}) \leq \lceil(1/\eta)^\beta\rceil\} \quad \text{(see eq. (H.31) for definition of the stopping time involved)}$$

and fix some $N > 0$. From Lemma H.6, we see that (for all sufficiently small $\eta$)

$$\mathbb{P}(A_1^\times(\eta)) \leq \eta^N.$$

Besides, using Lemma G.4 together with the fact that $\beta < \alpha(1 - \widetilde{\gamma})$, we know the existence of some constant $\theta > 0$ such that

$$\mathbb{P}(A_2^\times(\eta)) \leq \eta^\theta$$

for all sufficiently small $\eta$.

Now we focus on the behavior of the SGD iterates on event $\Big( A_1^\times(\eta) \cap A_2^\times(\eta) \Big)^c$. Let us arbitrarily choose some $x \in \Omega_i$ such that $|x - s_i| \vee |x - s_{i-1}| > 2\eta^\gamma$. First, from Lemma G.6 and eq. (H.32), we know that

$$|\mathbf{x}_t^\eta(x) - \mathbf{y}_{\lfloor t\rfloor}^\eta(x)| \leq 2\eta M \exp\big(2Mc_2\gamma\log(1/\eta)\big)$$
$$\leq 2\eta^{2\widetilde{\gamma}-2Mc_2\gamma} \leq 2\eta^{\widetilde{\gamma}} \leq \eta^\gamma \quad \forall t \leq 2c_2\gamma\log(1/\eta)/\eta \tag{H.33}$$

Next, from the definition of the function $t^\uparrow(\cdot)$ and eq. (H.33), we know that for

$$T_{\text{GD,return}}(x;\eta,\Delta) \triangleq \min\{n \geq 0 : \mathbf{y}_n^\eta(x) \in B(m_i, \frac{\Delta}{2} + \eta^\gamma)\}, \tag{H.34}$$

we have

$$T_{\text{GD,return}}(x;\eta,\Delta) \leq 2c_2\gamma\log(1/\eta)/\eta \tag{H.35}$$
$$\mathbf{y}_n^\eta > s_{i-1} + \eta^\gamma, \ \mathbf{y}_n^\eta < s_i - \eta^\gamma \ \forall n \leq 2c_2\gamma\log(1/\eta)/\eta. \tag{H.36}$$

Furthermore, on event $\Big( A_1^\times(\eta) \cap A_2^\times(\eta) \Big)^c$, due to Lemma G.5 and eq. (H.32), we have that

$$|X_n^\eta(x) - \mathbf{y}_n^\eta(x)| \leq \eta^{\widetilde{\gamma}/2} \exp\big(2Mc_2\log(1/\eta)\big) = \eta^{\frac{\widetilde{\gamma}}{2}-2Mc_2\gamma} \leq \eta^{\widetilde{\gamma}/4} < \eta^\gamma \ \forall n \leq 2c_2\gamma\log(1/\eta)/\eta.$$

Combining this with eq. (H.35), eq. (H.36), we can conclude that (recall that due to eq. (H.32) we have $2\eta^\gamma < \Delta/2$)

$$T_{\text{return}}^{(i)}(\eta,\Delta) \leq 2c_2\gamma\log(1/\eta)/\eta$$
$$X_n^\eta \in \Omega_i \ \forall n \leq 2c_2\gamma\log(1/\eta)/\eta$$

on event $\Big( A_1^\times(\eta) \cap A_2^\times(\eta) \Big)^c$. Therefore,

$$\liminf_{\eta\downarrow0} \inf_{x\in\Omega_i:|x-s_{i-1}|\vee|x-s_i|\geq2\eta^\gamma} \mathbb{P}_x\Big(T_{\text{return}}^{(i)}(\eta,\Delta) \leq \frac{2c_2\gamma\log(1/\eta)}{\eta}, \ X_n^\eta \in \Omega_i \ \forall n \leq T_{\text{return}}^{(i)}(\eta,\Delta)\Big)$$
$$\geq \liminf_{\eta\downarrow0} \mathbb{P}\Big(\Big(A_1^\times(\eta) \cap A_2^\times(\eta)\Big)^c\Big) \geq \liminf_{\eta\downarrow0} 1 - \eta^N - \eta^\theta = 1.$$

This concludes the proof. $\qquad\qquad\square$

The takeaway of the next lemma is that, almost always, the SGD iterates will quickly escape from the neighborhood of any $s_i$, the boundaries of each attraction fields.

**Lemma H.8.** *Given any $\gamma \in (0,1), t > 0$, we have*

$$\liminf_{\eta\downarrow0} \inf_{x\in[-L,L]} \mathbb{P}_x\Big( \min\{n \geq 0 : X_n^\eta \notin \bigcup_i B(s_i, 2\eta^\gamma)\} \leq \frac{t}{H(1/\eta)}\Big) = 1.$$

*Proof.* We only consider $\eta$ small enough so that

$$\min_{i=2,3,\cdots,n_{\min}-1} |s_i - s_{i-1}| > 3\eta^{\frac{1+\gamma}{2}},$$

$$\eta M < \eta^\gamma.$$

Also, the claim is trivial if $x \notin \cup_j B(s_j, 2\eta^\gamma)$, so without loss of generality we only consider the case where there is some $j \in [n_{\min}]$ and $x \in [-L, L], x \in B(s_j, 2\eta^\gamma)$. Let us define stopping times

$$T^\gamma \triangleq \min\{n \geq 1 : \eta|Z_n| > 5\eta^\gamma\}; \tag{H.37}$$

$$T^\gamma_{\text{escape}} \triangleq \min\{n \geq 0 : X^\eta_n \notin \cup_j B(s_j, 2\eta^\gamma)\}, \tag{H.38}$$

and the following two events

$$A^\times_1(\eta) \triangleq \{T^\gamma > \frac{t}{H(1/\eta)}\},$$

$$A^\times_2(\eta) \triangleq \{\eta|Z_{T^\gamma}| > \eta^{\frac{1+\gamma}{2}}\}.$$

First, using Lemma G.3 and the regularly varying nature of $H(\cdot) = \mathbb{P}(|Z_1| > \cdot)$, we know the existence of some $\theta > 0$ such that

$$\mathbb{P}(A^\times_1(\eta)) \leq \exp(-1/\eta^\theta)$$

for all $\eta > 0$ sufficiently small. Next, by definition of $T^\gamma$, one can see that (for any $\eta \in (0,1)$)

$$\mathbb{P}(A^\times_2(\eta)) = \frac{H(1/\eta^{\frac{1-\gamma}{2}})}{H(5/\eta^{1-\gamma})}.$$

Again, due to $H \in \mathcal{RV}_{-\alpha}$ and $1 - \gamma > 0$, we know the existence of some $\theta_1 > 0$ such that

$$\mathbb{P}(A^\times_2(\eta)) < \eta^{\theta_1}$$

for all $\eta > 0$ sufficiently small. To conclude the proof, we only need to note the following fact on event $\left(A^\times_1(\eta) \cup A^\times_2(\eta)\right)^c$. There are only two possibilities on this event: $T^\gamma_{\text{escape}} \leq T^\gamma - 1$, or $T^\gamma_{\text{escape}} \geq T^\gamma$. Now we analyze the two cases respectively.

- On $\left(A^\times_1(\eta) \cup A^\times_2(\eta)\right)^c \cap \{T^\gamma_{\text{escape}} \leq T^\gamma - 1\}$, we must have $T^\gamma_{\text{escape}} < T^\gamma \leq t/H(1/\eta)$.

- On $\left(A^\times_1(\eta) \cup A^\times_2(\eta)\right)^c \cap \{T^\gamma_{\text{escape}} \geq T^\gamma\}$, we know that at $n = T^\gamma - 1$, there exists an integer $j \in \{1, 2, \cdots, n_{\min} - 1\}$ such that $X^\eta_n \in B(s_j, 2\eta^\gamma)$. Now since $\eta M < \eta^\gamma$ and $\eta|Z_{T^\gamma}| > 5\eta^\gamma$, we must have

$$|X^\eta_{T^\gamma} - X^\eta_{T^\gamma-1}| > 4\eta^\gamma \Rightarrow X^\eta_{T^\gamma} \notin B(s_j, 2\eta^\gamma).$$

  On the other hand, the exclusion of event $A^\times_2(\eta)$ tells us that $|X^\eta_{T^\gamma} - X^\eta_{T^\gamma-1}| < 2\eta^{\frac{1+\gamma}{2}}$. Due to eq. (H.37), we then have $X^\eta_n \notin \cup_i B(s_i, 2\eta^\gamma)$.

In summary, $\left(A^\times_1(\eta) \cup A^\times_2(\eta)\right)^c \subseteq \{T^\gamma_{\text{return}} \leq t/H(1/\eta)\}$ and this conclude the proof. $\quad\square$

In the next lemma, we analyze the number of transitions needed to visit a certain local minimizer in the loss landscape. In general, we focus on a communication class $G$ and, for now, assume it is absorbing. Next, we introduce the following concepts to record the transitions between different local minimum. To be specific, for any $\eta > 0$ and any $\Delta > 0$ small enough so that $B(m_j, \Delta) \cap \Omega^c_j = \emptyset$ for all $j$, define

$$T_0(\eta, \Delta) = \min\{n \geq 0 : X^\eta_n \in \cup_j B(m_j, 2\Delta)\}; \tag{H.39}$$

$$I_0(\eta, \Delta) = j \text{ iff } X^\eta_{T_0(\eta,\Delta)} \in B(m_j, 2\Delta); \tag{H.40}$$

$$T_k(\eta, \Delta) = \min\{n > T_{k-1}(\eta, \Delta) : X^\eta_n \in \cup_{j \neq I_{k-1}(\eta,\Delta)} B(m_j, 2\Delta)\} \ \forall k \geq 1 \tag{H.41}$$

$$I_k(\eta, \Delta) = j \text{ iff } X^\eta_{T_k(\eta,\Delta)} \in B(m_j, 2\Delta) \ \forall k \geq 1. \tag{H.42}$$

As mentioned earlier, the next goal is to analyze the transitions between attraction fields it takes to visit $m_j$ when starting from $m_i$ when $m_i, m_j \in G$. Define

$$K_i(\eta, \Delta) \triangleq \min\{k \geq 0 : I_k(\eta, \Delta) = i\}.$$

**Lemma H.9.** *Assume that $G$ is an absorbing communication class on the graph $\mathcal{G}$. Then there exists some constant $p > 0$ such that for any $i$ with $m_i \in G$, any $\epsilon > 0$, and any $\Delta > 0$,*

$$\sup_{j:\, m_j \in G;\ x \in B(m_j, 2\Delta)} \mathbb{P}_x\big(K_i(\eta, \Delta) > u \cdot n_{min}\big) \leq \mathbb{P}\big(Geom(p) \geq u\big) + \epsilon \ \ \forall u = 1, 2, \cdots,$$

$$\sup_{j:\, m_j \in G;\ x \in B(m_j, 2\Delta)} \mathbb{P}_x\Big(\exists k \in [K_i(\eta, \Delta)] \ s.t. \ m_{I_{k}(\eta, \Delta)} \notin G\Big) \leq \epsilon$$

*hold for all $\eta > 0$ sufficiently small.*

*Proof.* The claim is trivial if, for the initial condition, we have $x \in B(m_i, 2\Delta)$. Next, let us observe the following facts.

- Define (recall the definitions of measure $\mu_i$ and sets $E_i, E_{i,j}$ in eq. (G.1)eq. (G.5)eq. (G.6))

$$J(j) \triangleq \arg\min_{\widetilde{j}:\, \mu_i(E_{j,\widetilde{j}}) > 0} |i - \widetilde{j}| \ \ \forall j \neq i$$

$$p^* \triangleq \min_{j:\, j \neq i,\ m_j \in G} \frac{\mu_j(E_{j, J(j)})}{\mu_j(E_j)}.$$

- From the definition of $J(j)$ and the fact that there are only finitely many attraction fields we can see that $p^* > 0$. Moreover, $G$ being a communication class implies that

$$|J(j) - i| < |j - i| \ \ \forall j \neq i,\ m_j \in G.$$

Indeed, if $i < j$, then since $G$ is a communication class and there are some $m_i \in G$ with $i < j$, we will at least have $\mu_j(E_{j, j-1}) > 0$, so $|J(j) - i| \leq |i - j| - 1$; the case that $i > j$ can be approached analogously.

- Now from the definition of $J(j)$ and Proposition G.25, together with the previous bullet point, we know that for all $\eta$ sufficiently small,

$$\inf_{x \in [-L, L]} \mathbb{P}_x\big(|I_{k+1} - i| \leq |I_k - i| - 1,\ m_{I_{k+1}} \in G \mid K_i(\eta, \Delta) > k,\ m_{I_k} \in G\big) \geq p^*/2$$

uniformly for all $k \geq 0$.

- Meanwhile, since $p^* > 0$, we are able to fix some $\delta > 0$ small enough such that

$$\frac{n_{min}\delta}{(p^*/2)^{n_{min}}} < \epsilon.$$

- On the other hand, for any $\widetilde{j}$ with $m_{\widetilde{j}} \notin G$, by definition of the typical transition graph we must have $\mu_j(E_{j,\widetilde{j}}) = 0$ for any $j$ with $m_j \in G$. Then due to Proposition G.25 again, one can see that for all $\eta > 0$ that is sufficiently small,

$$\sup_{x \in [-L, L]} \mathbb{P}_x\big(m_{I_{k+1}} \notin G \mid K_i(\eta, \Delta) > k,\ m_{I_k} \in G\big) < \delta$$

uniformly for all $k \geq 0$.

- Repeat this argument for $n_{min}$ times, and we can see that for all $\eta$ sufficiently small

$$\inf_{x \in [-L, L]} \mathbb{P}_x\big(K_i(\eta, \Delta) \leq k + n_{min} \mid K_i(\eta, \Delta) > k,\ m_{I_k} \in G\big) \geq \left(\frac{p^*}{2}\right)^{n_{min}}$$

$$\sup_{x \in [-L, L]} \mathbb{P}_x\big(\exists l \in [n_{min}] \ s.t. \ m_{I_{k+l}} \notin G \mid K_i(\eta, \Delta) > k,\ m_{I_k} \in G\big) \leq n_{min}\delta$$

uniformly for all $k \geq 1$.

- Lastly, to apply the bounds established above, we will make use of the following expression of several probabilities. For any $j$ with $m_j \in G$ and $x \in B(m_j, 2\Delta)$ and any $u = 1, 2, \cdots$,

$$\mathbb{P}_x(\exists l \in [un_{\min}] \text{ s.t. } m_{I_l} \notin G, \ K_i(\eta, \Delta) > u \cdot n_{\min})$$

$$= \sum_{v=0}^{u-1} \mathbb{P}_x\Big(\exists k \in [n_{\min}] \text{ s.t. } m_{I_{k+vn_{\min}}} \notin G \ \Big| \ K_i(\eta, \Delta) > v \cdot n_{\min}, \ m_{I_l} \in G \ \forall l \leq vn_{\min}\Big)$$

$$\cdot \prod_{w=0}^{v-1} \mathbb{P}_x\Big(K_i(\eta, \Delta) > (w+1)n_{\min}, \ m_{I_{k+wn_{\min}}} \in G \ \forall k \in [n_{\min}] \ \Big|$$

$$K_i(\eta, \Delta) > w \cdot n_{\min}, \ m_{I_l} \in G \ \forall l \leq wn_{\min}\Big)$$

$$\mathbb{P}_x(m_{I_l} \in G \ \forall l \in [un_{\min}], \ K_i(\eta, \Delta) > u \cdot n_{\min})$$

$$= \prod_{v=0}^{u-1} \mathbb{P}\Big(K_i(\eta, \Delta) > (v+1)n_{\min}, \ m_{I_{k+vn_{\min}}} \in G \ \forall k \in [n_{\min}] \ \Big|$$

$$K_i(\eta, \Delta) > vn_{\min}, \ m_{I_k} \in G \ \forall k \in [vn_{\min}]\Big)$$

In summary, now we can see that (for sufficiently small $\eta$)

$$\sup_{j: \, m_j \in G; \, x \in B(m_j, 2\Delta)} \mathbb{P}_x\Big(\exists k \in [K_i(\eta, \Delta)] \ s.t. \ m_{I_k(\eta, \Delta)} \notin G\Big)$$

$$\leq \sum_{u=0}^{\infty} \sup_{j: \, m_j \in G; \, x \in B(m_j, 2\Delta)} \mathbb{P}_x\Big(\exists v \in [n_{\min}] \text{ such that } m_{I_{v+un_{\min}}} \notin G,$$

$$K_i(\eta, \Delta) > un_{\min}, m_{I_k} \in G \ \forall k \in [un_{\min}]\Big)$$

$$\leq \sum_{u=0}^{\infty} \sup_{j: \, m_j \in G; \, x \in B(m_j, 2\Delta)} \mathbb{P}_x\Big(\exists v \in [n_{\min}] \text{ such that } m_{I_{v+un_{\min}}} \notin G \ \Big|$$

$$K_i(\eta, \Delta) > un_{\min}, m_{I_k} \in G \ \forall k \in [un_{\min}]\Big)$$

$$\cdot \prod_{v=0}^{u-1} \sup_{j: \, m_j \in G; \, x \in B(m_j, 2\Delta)} \mathbb{P}_x\Big(K_i(\eta, \Delta) > (v+1)n_{\min}, m_{I_{k+vn_{\min}}} \in G \ \forall k \in [n_{\min}] \Big|$$

$$K_i(\eta, \Delta) > vn_{\min}, m_{I_k} \in G \ \forall k \in [vn_{\min}]\Big)$$

$$\leq \sum_{u \geq 0} n_{\min} \delta \Big(1 - (\frac{p^*}{2})^{n_{\min}}\Big)^{u-1} = \frac{n_{\min} \delta}{(p^*/2)^{n_{\min}}} \leq \epsilon$$

and

$$\sup_{j: m_j \in G, \, x \in B(m_j, 2\Delta)} \mathbb{P}_x(K_i(\eta, \Delta) > u \cdot n_{\min})$$

$$\leq \sup_{j: m_j \in G, \, x \in B(m_j, 2\Delta)} \mathbb{P}_x(\exists l \in [un_{\min}] \text{ s.t. } m_{I_l} \notin G, \ K_i(\eta, \Delta) > u \cdot n_{\min})$$

$$+ \sup_{j: m_j \in G, \, x \in B(m_j, 2\Delta)} \mathbb{P}_x(m_{I_l} \in G \ \forall l \in [un_{\min}], \ K_i(\eta, \Delta) > u \cdot n_{\min})$$

$$\leq \sum_{v=1}^{u} n_{\min} \delta \Big(1 - (\frac{p^*}{2})^{n_{\min}}\Big)^{v-1} + \Big(1 - (\frac{p^*}{2})^{n_{\min}}\Big)^{u}$$

$$\leq \frac{n_{\min} \delta}{(p^*/2)^{n_{\min}}} + \Big(1 - (\frac{p^*}{2})^{n_{\min}}\Big)^{u}$$

$$\leq \epsilon + \Big(1 - (\frac{p^*}{2})^{n_{\min}}\Big)^{u}$$

uniformly for all $u = 1, 2, \cdots$. To conclude the proof, it suffices to set $p = (\frac{p^*}{2})^{n_{\min}}$. $\qquad\square$

The proof above can be easily adapted to the case when the communication class $G$ is transient. Define

$$K_i^G(\eta, \Delta) \triangleq \min\{k \geq 0 : I_k(\eta, \Delta) = i \text{ or } m_{I_k(\eta, \Delta)} \notin G\}.$$

**Lemma H.10.** *Assume that $G$ is a transient communication class on the graph $\mathcal{G}$. Then there exists some constant $p > 0$ such that for any $i$ with $m_i \in G$ and any $\Delta \in (0, \bar{\epsilon}/3)$,*

$$\sup_{j:\, m_j \in G;\, x \in B(m_j, 2\Delta)} \mathbb{P}_x(K_i^G(\eta, \Delta) > u \cdot n_{min}) \leq \mathbb{P}(Geom(p) \geq u) \;\; \forall u = 1, 2, \cdots \qquad \text{(H.43)}$$

*hold for all $\eta > 0$ sufficiently small.*

*Proof.* The structure of this proof is analogous to that of Lemma H.9. Again, the claim is trivial if, for the initial condition, we have $x \in B(m_i, 2\Delta)$. Next, let us observe the following facts.

- Define (recall the definitions of measure $\mu_i$ and sets $E_i$, $E_{i,j}$ in eq. (G.1)eq. (G.5)eq. (G.6))

$$J(j) \triangleq \arg \min_{\widetilde{j}:\mu_i(E_{j,\widetilde{j}})>0} |i - \widetilde{j}| \;\; \forall j \neq i$$

$$p^* \triangleq \min_{j:\, j \neq i,\, m_j \in G} \frac{\mu_j(E_{j,J(j)})}{\mu_j(E_j)}.$$

- From the definition of $J(j)$ and the fact that there are only finitely many attraction fields we can see that $p^* > 0$. Moreover, $G$ being a communication class implies that

$$|J(j) - i| < |j - i| \;\; \forall j \neq i,\, m_j \in G.$$

  Indeed, if $i < j$, then since $G$ is a communication class and there are some $m_i \in G$ with $i < j$, we will at least have $\mu_j(E_{j,j-1}) > 0$, so $|J(j) - i| \leq |i - j| - 1$; the case that $i > j$ can be approached analogously.

- Now from the definition of $J(j)$ and Proposition G.25, together with the previous bullet point, we know that for all $\eta$ sufficiently small,

$$\inf_{x \in [-L, L]} \mathbb{P}_x(|I_{k+1} - i| \leq |I_k - i| - 1,\, m_{I_{k+1}} \in G \mid K_i^G(\eta, \Delta) > k) \geq p^*/2$$

  uniformly for all $k \geq 0$.

- Repeat this argument for $n_{min}$ times, and we can see that for all $\eta$ sufficiently small

$$\inf_{x \in [-L, L]} \mathbb{P}_x(K_i^G(\eta, \Delta) \leq k + n_{min} \mid K_i^G(\eta, \Delta) > k) \geq \left(\frac{p^*}{2}\right)^{n_{min}}$$

  uniformly for all $k \geq 1$.

- Lastly, for any $j \neq i$ with $m_j \in G$ and $x \in B(m_j, 2\Delta)$ and any $u = 1, 2, \cdots$,

$$\mathbb{P}_x(K_i^G(\eta, \Delta) > u \cdot n_{min})$$

$$= \prod_{v=0}^{u-1} \mathbb{P}_x\left(K_i^G(\eta, \Delta) > (v+1)n_{min} \;\middle|\; K_i^G(\eta, \Delta) > v \cdot n_{min}\right)$$

$$= \prod_{v=0}^{u-1} \left(1 - \mathbb{P}_x\left(K_i^G(\eta, \Delta) \leq (v+1)n_{min} \;\middle|\; K_i^G(\eta, \Delta) > v \cdot n_{min}\right)\right)$$

In summary, now we can see that (for sufficiently small $\eta$)

$$\sup_{j:m_j \in G,\, x \in B(m_j, 2\Delta)} \mathbb{P}_x(K_i^G(\eta, \Delta) \geq u \cdot n_{min}) \leq \left(1 - (\frac{p^*}{2})^{n_{min}}\right)^u$$

uniformly for all $u = 1, 2, \cdots$. To conclude the proof, it suffices to set $p = (\frac{p^*}{2})^{n_{min}}$. $\qquad \square$

We are now ready to prove Lemma H.1, which, as demonstrated earlier, is the key tool in proof of Theorem 2.

*Proof of Lemma H.1.* The claim is trivial if $l^{\text{large}} = 1$, so we focus on the case where $l^{\text{large}} \geq 2$. Fix some

$$\widetilde{\gamma} \in (0, (1 - \frac{1}{\alpha}) \wedge (\frac{1}{2})), \ \beta \in \big(1, (2 - 2\widetilde{\gamma}) \wedge (\alpha - \alpha\widetilde{\gamma})\big), \ \gamma \in (0, \frac{\widetilde{\gamma}}{16Mc_2} \wedge \frac{\widetilde{\gamma}}{4}).$$

Let $q^* = \max_j \mu_j(E_j(0))$. We show that for any $t \in (0, \frac{\delta}{4q^*})$ the claim is true.

Now we only consider $\Delta \in (0, \bar{\epsilon}/3)$ and $\eta$ small enough so that $\eta M \leq \eta^{\gamma}$ and $\eta^{\gamma} < \Delta$. Consider the following stopping times

$$T^{\gamma}_{\text{escape}} \triangleq \min\{n \geq 0 : X^{\eta}_n \notin \cup_j B(s_j, 2\eta^{\gamma})\};$$
$$T^{\gamma}_{\text{return}} \triangleq \min\{n \geq 0 : X^{\eta}_n \in \cup_j B(m_j, 2\eta^{\gamma})\}.$$

First, from Lemma H.8, we know that

$$\sup_{x \in [-L,L]} \mathbb{P}_x(T^{\gamma}_{\text{escape}} > 1/H(1/\eta)) < \delta/2$$

for all $\eta$ sufficiently small. Besides, by combining Lemma H.7 with Markov property (applied at $T^{\gamma}_{\text{escape}}$), we have

$$\sup_{x \in [-L,L]} \mathbb{P}_x\Big( T^{\gamma}_{\text{return}} - T^{\gamma}_{\text{escape}} > 2c_2\gamma \log(1/\eta)/\eta \ \Big| \ T^{\gamma}_{\text{escape}} \leq \frac{1}{H(1/\eta)}\Big) < \delta/2$$

for all $\eta$ sufficiently small. Therefore, for all $\eta$ sufficiently small,

$$\sup_{x \in [-L,L]} \mathbb{P}\Big(T^{\gamma}_{\text{return}} > \frac{1}{H(1/\eta)} + 2c_2\gamma\frac{\log(1/\eta)}{\eta}\Big) < \delta. \tag{H.44}$$

Let $J$ be the unique index such that $X^{\eta}_{T^{\gamma}_{\text{return}}} \in \Omega_J$. Our next goal is to show that, almost always, the SGD iterates will visit the local minimum at some *large* attraction fields. Therefore, without loss of generality, we can assume that $m_J \notin M^{\text{large}}$, and define

$$T^{\gamma}_{\text{large}} \triangleq \min\{n \geq T^{\gamma}_{\text{return}} : \ X^{\eta}_n \in \bigcup_{i:m_i \in M^{\text{large}}} B(m_i, 2\Delta)\}$$

and introduce the following definitions:

$$\tau_0 \triangleq T^{\gamma}_{\text{return}}, \ J_0 \triangleq J$$
$$\tau_k \triangleq \min\{n > \tau_{k-1} : \ X^{\eta}_n \in \bigcup_{j \neq J_{k-1}} B(m_j, 2\Delta)\}$$
$$J_k = j \Leftrightarrow X^{\eta}_{\tau_k} \in \Omega_j \ \forall k \geq 1$$
$$K \triangleq \min\{k \geq 0 : \ m_{J_k} \in M^{\text{large}}\}.$$

In other words, the sequence of stopping times $(\tau_k)_{k \geq 1}$ is the time that, starting from $T^{\gamma}_{\text{return}}$, the SGD iterates visited a local minimum that is different from the one visited at $\tau_{k-1}$, and $(J_k)_{k \geq 0}$ records the label of the visited local minima. The random variable $K$ is the number of transitions required to visit a minimizer in a *large* attraction field. From Lemma H.9, we know the existence of some $p^* > 0$ such that (for all $\eta$ sufficiently small)

$$\sup_{x \in [-L,L]} \mathbb{P}_x(K \geq u \cdot n_{\min}) \leq \mathbb{P}\Big(\text{Geom}(p^*) \geq u\Big) + \frac{\delta}{2} \ \forall u = 1, 2, 3, \cdots.$$

where $\text{Geom}(a)$ is a Geometric random variable with success rate $a \in (0, 1)$. Therefore, one can find integer $N(\delta)$ such that (for all sufficiently small $\eta$)

$$\sup_{x \in [-L,L]} \mathbb{P}(K \geq N(\delta)) \leq \delta. \tag{H.45}$$

Next, given results in Proposition G.25 and the fact that there are only finitely many attraction fields, one can find a real number $u(\delta)$ such that (for all sufficiently small $\eta$)

$$\sup_{x \in [-L,L]} \mathbb{P}_x(\tau_k - \tau_{k-1} \leq \frac{u(\delta)}{\lambda_{J_{k-1}}(\eta)}) \leq \delta/N(\delta) \tag{H.46}$$

uniformly for all $k = 1, 2, \cdots, N(\delta)$. From eq. (H.44), eq. (H.45), eq. (H.46), we now have

$$\sup_{x \in [-L,L]} \mathbb{P}_x\Big(X_n^\eta \notin \bigcup_{j: m_j \in M^{\text{large}}} B(m_j, 2\Delta) \ \forall n \leq N(\delta)u(\delta)\frac{H(1/\eta)/\eta}{\lambda^{\text{large}}(\eta)} + \frac{1}{H(1/\eta)} + 2c_2\gamma\frac{\log(1/\eta)}{\eta}\Big)$$

$$\leq 3\delta \tag{H.47}$$

for any sufficiently small $\eta$. To conclude the proof we just observe the following facts. First, due to $H \in \mathcal{RV}_{-\alpha}$ and $l^{\text{large}} \geq 2$, we have

$$\lim_{\eta \downarrow 0} H(1/\eta)/\eta = 0, \ \lim_{\eta \downarrow 0} \frac{\lambda^{\text{large}}(\eta)}{H(1/\eta)} = 0, \ \lim_{\eta \downarrow 0} \frac{\log(1/\eta)}{\eta}\lambda^{\text{large}}(\eta) = 0.$$

Therefore, for sufficiently small $\eta$, we will have (note that $\epsilon, \delta$ are fixed constants in this proof, so $N(\delta), u(\delta)$ are also fixed)

$$\frac{N(\delta)u(\delta)\frac{H(1/\eta)/\eta}{\lambda^{\text{large}}(\eta)} + \frac{1}{H(1/\eta)} + 2c_2\gamma\frac{\log(1/\eta)}{\eta}}{\lfloor t/\lambda^{\text{large}}(\eta) \rfloor} \leq \epsilon. \tag{H.48}$$

Second, recall that we fixed some $t \in (0, \frac{\delta}{4q^*})$ where $q^* = \max_j \mu_j(E_j)$. Also, choose some $C > 0$ small enough so that

$$C < \delta/2, \ 2(1+C)^2 < 4.$$

From Proposition G.23 and the fact that there are only finitely many attraction fields, there exists some $\bar{\eta}_0 > 0$ such that for any $\eta \in (0, \bar{\eta}_0)$ and any $\Delta > 0$ sufficiently small,

$$\sup_{i: m_i \in M^{\text{large}}} \sup_{x \in [m_i - 2\Delta, m_i + 2\Delta]} \mathbb{P}_x\Big(\sigma_i(\eta) \leq \frac{t}{\lambda^{\text{large}}(\eta)}\Big)$$

$$\leq \sup_{i: m_i \in M^{\text{large}}} \sup_{x \in [m_i - 2\Delta, m_i + 2\Delta]} \mathbb{P}_x\Big(\mu_i(E_i)\lambda^{\text{large}}(\eta)\sigma_i(\eta) \leq q^*t\Big)$$

$$\leq C + 2(1+C)^2 q^*t \leq 2\delta.$$

Combine this bound with Markov property (applied at $\tau_K$), and we obtain that

$$\sup_{x \in [-L,L]} \mathbb{P}_x\Big(\exists n \in \big[\lfloor t/\lambda^{\text{large}}(\eta) \rfloor\big] \ s.t. \ X_{n+\tau_K}^\eta \notin \bigcup_{i: m_i \in M^{\text{large}}} \Omega_i\Big) \leq 2\delta$$

for all $\eta$ sufficiently small. Together with eq. (H.47), eq. (H.48), we have shown that

$$\sup_{x \in [-L,L]} \mathbb{P}_x\Big(V^{\text{small}}(\eta, \epsilon, t) > \epsilon\Big) \leq 5\delta$$

holds for all $\eta$ sufficiently small. $\qquad \square$

## H.2 Proof of Lemma H.4, H.5

We shall return to the discussion about the dynamics of SGD iterates on a communication class $G$. Recall that

$$G^{\text{large}} = \{m_1^{\text{large}}, \cdots, m_{i_G}^{\text{large}}\}, \ G^{\text{small}} = \{m_1^{\text{small}}, \cdots, m_{i_G'}^{\text{small}}\}.$$

If $X_n^\eta$ is initialized at some sharp minimum on $G$, then we are interested in the behavior of $X_n^\eta$ at the first visit to some large attraction fields on $G$. Define

$$T_G(\eta, \Delta) \triangleq \min\{n \geq 0: X_n^\eta \in \bigcup_{i: \ m_i \in G^{\text{large}}} B(m_i, 2\Delta) \text{ or } X_n^\eta \notin \cup_{i: \ m_i \in G} \Omega_i\}. \tag{H.49}$$

Not only is this definition of $T_G$ analogous to the one for $T_G^{DTMC}$ in eq. (H.16), but, as illustrated in the next lemma, $T_G$ also behaves similarly as $T_G$ on a communication class $G$ in the following sense: the probabilities $p_{i,j}$ defined in eq. (H.17) govern the dynamics regarding which large attraction field on $G$ is the first one to be visited. Besides, $T_G$ is usually rather small, meaning that the SGD iterates would efficiently arrive at a large attraction field on $G$ or simply escape from $G$.

**Lemma H.11.** *Given any $\theta \in (0, (\alpha - 1)/2)$, $\epsilon \in (0, 1)$, $i, j \in [n_{min}]$ such that $m_i \in G^{small}$, $m_j \notin G^{large}$, the following claims hold for all $\Delta > 0$ that is sufficiently small:*

$$\limsup_{\eta \downarrow 0} \sup_{x \in B(m_i, 2\Delta)} \mathbb{P}_x \left( T_G(\eta, \Delta) \leq \frac{\eta^\theta}{\lambda_G(\eta)}, \, X^\eta_{T_G} \in B(m_j, 2\Delta) \right) \leq p_{i,j} + 5\epsilon,$$

$$\liminf_{\eta \downarrow 0} \inf_{x \in B(m_i, 2\Delta)} \mathbb{P}_x \left( T_G(\eta, \Delta) \leq \frac{\eta^\theta}{\lambda_G(\eta)}, \, X^\eta_{T_G} \in B(m_j, 2\Delta) \right) \geq p_{i,j} - 5\epsilon,$$

$$\limsup_{\eta \downarrow 0} \sup_{x \in B(m_i, 2\Delta)} \mathbb{P}_x \left( T_G(\eta, \Delta) > \frac{\eta^\theta}{\lambda_G(\eta)} \right) \leq 2\epsilon.$$

*Proof.* For $G^{small} \neq \emptyset$ to hold (and the discussion to be meaningful), we must have $l^*_G \geq 2$. Throughout the proof, we assume this is the case. Besides, we require that $\Delta \in (0, \bar{\epsilon}/3)$ so we have

$$B(m_i, 3\Delta) \cap \Omega^c_i = \emptyset \; \forall i \in [n_{min}]$$

and the $3\Delta$-neighborhood of each local minimum will not intersect with each other. In this proof we will only consider $\Delta$ in this range.

From Lemma H.9 (if $G$ is absorbing) or Lemma H.10 (if $G$ is transient), we know the existence of some integer $N(\epsilon)$ such that for (see the definition of $I_k$ in eq. (H.39)-eq. (H.42))

$$N_G(\eta, \Delta) \triangleq \min\{k \geq 0 : \, m_{I_k(\eta, \Delta)} \in G^{large} \text{ or } m_{I_k(\eta, \Delta)} \notin G\},$$

we have

$$\sup_{x \in B(m_i, 2\Delta)} \mathbb{P}_x \left( N_G(\eta, \Delta) > N(\epsilon) \right) < \epsilon$$

for all $\eta$ sufficiently small. Fix such $N(\epsilon)$. Next, from Proposition G.25, we can find $u(\epsilon) \in (0, \infty)$ and $\bar{\Delta} \in (0, \bar{\epsilon}/3)$ such that for all $\Delta \in (0, \bar{\Delta})$, we have

$$\sup_{x \in B(m_i, 2\Delta)} \mathbb{P}_x \left( T_k(\eta, \Delta) - T_{k-1}(\eta, \Delta) > u(\epsilon)/\Lambda \big( I_{k-1}(\eta, \Delta), \eta \big) \right) \leq \epsilon/N(\epsilon) \; \forall k \in [N(\epsilon)]$$

for all $\eta$ sufficiently small. Fix such $u(\epsilon)$ and $\bar{\Delta}$. Now note that on the event

$$A \triangleq \left\{ N_G \leq N(\epsilon) \right\} \cap \left\{ T_k(\eta, \Delta) - T_{k-1}(\eta, \Delta) \leq u(\epsilon)/\Lambda \big( I_{k-1}(\eta, \Delta), \eta \big) \; \forall k \in [N(\epsilon)] \right\},$$

due to the choice of $\theta \in (0, (\alpha - 1)/2)$ and $H \in \mathcal{RV}_{-\alpha}$, we have (when $\eta \in (0, 1)$)

$$T_k(\eta, \Delta) - T_{k-1}(\eta, \Delta) \leq \frac{\eta^{2\theta}}{\lambda_G(\eta)} \; \forall k < N_G(\eta, \Delta)$$

$$\Rightarrow T_G(\eta, \Delta) = T_{N_G(\eta, \Delta)}(\eta, \Delta) \leq N(\epsilon) u(\epsilon) \frac{\eta^{2\theta}}{\lambda_G(\eta)}.$$

For any $\eta$ sufficiently small, we will have $N(\epsilon) u(\epsilon) \frac{\eta^{2\theta}}{\lambda_G(\eta)} < \frac{\eta^\theta}{\lambda_G(\eta)}$. In summary, we have established that for all $\Delta \in (0, \bar{\Delta})$,

$$\limsup_{\eta \downarrow 0} \sup_{x \in B(m_i, 2\Delta)} \mathbb{P}_x \left( T_G > \frac{\eta^\theta}{\lambda_G(\eta)} \right) \leq \limsup_{\eta \downarrow 0} \sup_{x \in B(m_i, 2\Delta)} \mathbb{P}_x (A^c) < 2\epsilon. \tag{H.50}$$

Next, let

$$\mathbf{S}(\epsilon) \triangleq \left\{ (m'_1, \cdots, m'_{N(\epsilon)}) \in \{m_1, \cdots, m_{n_{min}}\}^{N(\epsilon)} : \, \exists k \in [N(\epsilon)] \text{ s.t. } m'_k = m_j \right\}.$$

We can see that $\mathbf{S}(\epsilon)$ contains all the possible transition path for $Y^{DTMC}$ where the state $m_j$ is visited within the first $N(\epsilon)$ steps. Obviously, $|\mathbf{S}(\epsilon)| < \infty$. Let $\epsilon_1 = \epsilon/|\mathbf{S}(\epsilon)|$. If we are able to show the existence of some $\bar{\Delta}_1 > 0$ such that for all $\Delta \in (0, \bar{\Delta}_1)$, the following claim holds for any $(m'_k)^{N(\epsilon)}_{k=1} \in \mathbf{S}(\epsilon)$:

$$\limsup_{\eta \downarrow 0} \sup_{x \in B(m_i, 2\Delta)} \left| \mathbb{P}_x \left( m_{I_k} = m'_k \; \forall k \in [N(\epsilon)] \right) - \mathbb{P} \left( Y^{DTMC}_k(m_i) = m'_k \; \forall k \in [N(\epsilon)] \right) \right| < \epsilon_1, \tag{H.51}$$

then we must have (for all $\Delta \in (0, \bar{\Delta} \wedge \bar{\Delta}_1)$)

$$\limsup_{\eta \downarrow 0} \sup_{x \in B(m_i, 2\Delta)} \left| \mathbb{P}_x \left( X^\eta_{T_G} \in B(m_j, 2\Delta) \right) - p_{i,j} \right|$$

$$= \limsup_{\eta \downarrow 0} \sup_{x \in B(m_i, 2\Delta)} \left| \mathbb{P}_x \left( X^\eta_{T_G} \in B(m_j, 2\Delta), T_G \leq N(\epsilon) \right) + \mathbb{P}_x \left( X^\eta_{T_G} \in B(m_j, 2\Delta), T_G > N(\epsilon) \right) \right.$$

$$\left. - \mathbb{P} \left( Y^{DTMC}_{T_G^{DTMC}}(m_i) = m_j, T_G^{DTMC} \leq N(\epsilon) \right) - \mathbb{P} \left( Y^{DTMC}_{T_G^{DTMC}}(m_i) = m_j, T_G^{DTMC} > N(\epsilon) \right) \right|$$

$$\leq \limsup_{\eta \downarrow 0} \sup_{x \in B(m_i, 2\Delta)} \left| \mathbb{P}_x \left( X^\eta_{T_G} \in B(m_j, 2\Delta), T_G \leq N(\epsilon) \right) \right.$$

$$\left. - \mathbb{P} \left( Y^{DTMC}_{T_G^{DTMC}}(m_i) = m_j, T_G^{DTMC} \leq N(\epsilon) \right) \right|$$

$$+ \limsup_{\eta \downarrow 0} \sup_{x \in B(m_i, 2\Delta)} \mathbb{P}_x(T_G > N(\epsilon)) + \mathbb{P}(T_G^{DTMC}(m_i) > N(\epsilon))$$

$$\leq |\mathbf{S}(\epsilon)| \epsilon_1 + \limsup_{\eta \downarrow 0} \sup_{x \in B(m_i, 2\Delta)} \mathbb{P}_x(T_G > N(\epsilon)) + \mathbb{P}(T_G^{DTMC}(m_i) > N(\epsilon))$$

$$\leq 3\epsilon.$$

To show that eq. (H.51) is true, we fix some $(m'_k)_{k=1}^{N(\epsilon)} \in \mathbf{S}(\epsilon)$ and let $(\mathbf{k}'(k))_{k=1}^{N(\epsilon)}$ be the sequence with $m_{\mathbf{k}'(k)} = m'_k$ for each $k \in [N(\epsilon)]$. From the definition of $Y^{DTMC}$ we have (let $\mathbf{k}'(0) = i$)

$$\mathbb{P} \left( Y^{DTMC}_k(m_i) = m'_k \ \forall k \in [N(\epsilon)] \right) = \prod_{k=0}^{N(\epsilon)-1} \frac{\mu_{\mathbf{k}'(k)}(E_{\mathbf{k}'(k), \mathbf{k}'(k+1)})}{\mu_{\mathbf{k}'(k)}(E_{\mathbf{k}'(k)})}.$$

On the other hand, using Proposition G.25, we know that for any arbitrarily chosen $\epsilon' \in (0, 1)$, we have

$$\limsup_{\eta \downarrow 0} \sup_{x \in B(m_i, 2\Delta)} \mathbb{P}_x \left( m_{I_k(\eta, \Delta)} = m'_k \ \forall k \in [N(\epsilon)] \right) \leq \prod_{k=0}^{N(\epsilon)-1} \frac{\mu_{\mathbf{k}'(k)}(E_{\mathbf{k}'(k), \mathbf{k}'(k+1)})}{\mu_{\mathbf{k}'(k)}(E_{\mathbf{k}'(k)})} \cdot (1 + \epsilon'),$$

$$\liminf_{\eta \downarrow 0} \inf_{x \in B(m_i, 2\Delta)} \mathbb{P}_x \left( m_{I_k(\eta, \Delta)} = m'_k \ \forall k \in [N(\epsilon)] \right) \geq \prod_{k=0}^{N(\epsilon)-1} \frac{\mu_{\mathbf{k}'(k)}(E_{\mathbf{k}'(k), \mathbf{k}'(k+1)})}{\mu_{\mathbf{k}'(k)}(E_{\mathbf{k}'(k)})} \cdot (1 - \epsilon'),$$

for all $\Delta > 0$ sufficiently small. The arbitrariness of $\epsilon' > 0$, together with $|\mathbf{S}(\epsilon)| < \infty$, allows us to see the existence of some $\bar{\Delta}_1 > 0$ such that with $\Delta \in (0, \bar{\Delta}_1)$, eq. (H.51) holds for any $(m'_k)_{k=1}^{N(\epsilon)} \in \mathbf{S}(\epsilon)$. To conclude the proof, observe that

$$\limsup_{\eta \downarrow 0} \sup_{x \in B(m_i, 2\Delta)} \left| \mathbb{P}_x \left( X^\eta_{T_G} \in B(m_j, 2\Delta) \right) - \mathbb{P}_x \left( T_G(\eta, \Delta) \leq \frac{\eta^\theta}{\lambda_G(\eta)}, X^\eta_{T_G} \in B(m_j, 2\Delta) \right) \right|$$

$$\leq \limsup_{\eta \downarrow 0} \sup_{x \in B(m_i, 2\Delta)} \mathbb{P}_x \left( T_G > \frac{\eta^\theta}{\lambda_G(\eta)} \right) < \epsilon$$

due to eq. (H.50). $\qquad \square$

Recall that continuous-time process $X^{*, \eta}$ is the *scaled* version of $X^\eta$ defined in eq. (H.13), and the mapping $\mathbf{T}^*(n, \eta) \triangleq n\lambda_G(\eta)$ returns the timestamp $t$ for $X^{*, \eta}_t$ corresponding to the unscaled step $n$ on the time horizon of $X^\eta_n$. As an *inverse* mapping of $\mathbf{T}^*$, we define the mapping $\mathbf{N}^*(t, \eta) = \lfloor t/\lambda_G(\eta) \rfloor$ that maps the scaled timestamp $t$ back to the step number $n$ for the unscaled process $X^\eta$.

In the next lemma, we show that, provided that $X^{*, \eta}$ stays on a communication class $G$ before some time $t$, the scaled process $X^{*, \eta}_t$ is almost always in the largest attraction fields of a communication class $G$.

**Lemma H.12.** *Let $G$ be a communication class on the graph $\mathcal{G}$. Given any $\epsilon_1 > 0$, $t > 0$ and any $x \in \Omega_i$ with $m_i \in G$, the following claim holds for all $\Delta > 0$ small enough:*

$$\limsup_{\eta \downarrow 0} \mathbb{P}_x \left( \left\{ X^{*, \eta}_t \notin \bigcup_{j: m_j \in G^{large}} B(m_j, 3\Delta) \right\} \cap \left\{ X^{*, \eta}_s \in \bigcup_{k: m_k \in G} \Omega_k \ \forall s \in [0, t] \right\} \right) \leq 2\epsilon_1.$$

*Proof.* Let $\Delta \in (0, \bar{\epsilon}/3)$ for the constant $\bar{\epsilon}$ in eq. (G.28)eq. (G.29), so we are certain that each $B(m_i, 2\Delta)$ lies entirely in $\Omega_i$ and would not intersect with each other since

$$B(m_i, 3\Delta) \cap \Omega_i^c = \emptyset \ \ \forall i \in [n_{\min}].$$

Besides, with $\epsilon = \Delta/3$, we know the existence of some $\delta > 0$ such that claims in Lemma G.14 would hold of the chosen $\epsilon, \delta$. Fix such $\delta$ for the entirety of this proof. Lastly, fix some

$$\widetilde{\gamma} \in (0, (1 - \frac{1}{\alpha}) \wedge (\frac{1}{2})), \ \beta \in \left(1, (2 - 2\widetilde{\gamma}) \wedge (\alpha - \alpha\widetilde{\gamma})\right), \ \gamma \in (0, \frac{\widetilde{\gamma}}{16Mc_2} \wedge \frac{\widetilde{\gamma}}{4}).$$

The blueprint of this proof is as follows. We will define a sequence of stopping times $(N_j)_{j=1}^6$ such that the corresponding scaled timestamps $\mathbf{T}_j^* = \mathbf{T}^*(N_j, \eta)$ gradually approach $t$. By analyzing the behavior of $X^{*,\eta}$ on a time interval $[t - \Delta_t, t]$ that is very close to $t$ (in particular, on the aforementioned stopping times $\mathbf{T}_j^*$), we are able to establish the properties of a series of events $A_1 \supseteq A_2 \supseteq A_3$. Moreover, we will show that $A_3 \subseteq \{X_t^{*,\eta,\Delta} \in \bigcup_{i:\ m_i \in G^{\text{large}}} B(m_i, 3\Delta)\}$, so the properties about events $A, A_2, A_3$ can be used to bound the probability of the target event.

Arbitrarily choose some $\Delta_t \in (0, t)$. To proceed, let $N_0 \triangleq \mathbf{N}^*(t - \Delta_t, \eta)$ be the stopping time corresponding to timestamp $t - \Delta_t$ for the scaled process. Using Lemma H.8, we know that for stopping time $N_1 \triangleq \min\{n \geq N_0 : X_n^\eta \notin \cup_j B(s_j, 2\eta^\gamma)\}$, we have

$$\liminf_{\eta \downarrow 0} \inf_{x \in [-L, L]} \mathbb{P}_x(N_1 - N_0 < \frac{\Delta_t/4}{H(1/\eta)}) = 1.$$

Next, let $N_2 \triangleq \min\{n \geq N_1 : X_n^\eta \in \cup_j B(m_j, 2\Delta)\}$. From Lemma H.7 and $H \in \mathcal{RV}_{-\alpha}$ (so that $\log(1/\eta)/\eta = o(H(1/\eta))$), we have

$$\liminf_{\eta \downarrow 0} \inf_{x \in [-L, L]} \mathbb{P}_x(N_2 - N_1 < \frac{\Delta_t/4}{H(1/\eta)}) = 1.$$

Collecting results above, we have

$$\liminf_{\eta \downarrow 0} \inf_{x \in [-L, L]} \mathbb{P}_x(N_2 - N_0 < \frac{\Delta_t/2}{H(1/\eta)}) = 1. \tag{H.52}$$

Now note the following fact on the event $\{N_2 - N_0 < \frac{\Delta_t/2}{H(1/\eta)}\}$. The definition of the mapping $\mathbf{T}^*$ implies that, for any pair of positive integers $n_1 \leq n_2$, we have $\mathbf{T}^*(n_2, \eta) - \mathbf{T}^*(n_1, \eta) = (n_2 - n_1)\lambda_G(\eta) \leq (n_2 - n_1) \cdot H(1/\eta)$. Therefore, on $\{N_2 - N_0 \leq \frac{\Delta_t/2}{H(1/\eta)}\}$ we have

$$\mathbf{T}^*(N_2, \eta) - \mathbf{T}^*(N_0, \eta) < \Delta_t/2 \Rightarrow \mathbf{T}^*(N_2, \eta) < t - \frac{\Delta_t}{2}.$$

Besides, let $\mathbf{T}_2^* = \mathbf{T}^*(N_2, \eta)$. Now we can see that for event

$$A_0 \triangleq \left\{X_s^{*,\eta} \in \bigcup_{k:\ m_k \in G} \Omega_k \ \forall s \in [0, t]\right\} \cap \left\{N_2 - N_0 < \frac{\Delta_t/2}{H(1/\eta)}\right\},$$

we have

$$A_0 \subseteq A_1 \triangleq \left\{\mathbf{T}_2^* < t - \frac{\Delta_t}{2}\right\} \cap \left\{X_s^{*,\eta} \in \bigcup_{k:\ m_k \in G} \Omega_k \ \forall s \in [0, \mathbf{T}_2^*]\right\}.$$

Meanwhile, from eq. (H.52) we obtain that

$$\limsup_{\eta \downarrow 0} \sup_{x \in [-L, L]} \mathbb{P}_x\left(A_1^c \cap \{X_s^{*,\eta} \in \bigcup_{k:\ m_k \in G} \Omega_k \ \forall s \in [0, t]\}\right) = 0. \tag{H.53}$$

Moving on, we consider the following stopping times

$$N_3 \triangleq \min\{n \geq N_2 : X_n^\eta \in \bigcup_{j:\ m_j \in G^{\text{large}}} B(m_j, 2\Delta) \text{ or } X_n^\eta \notin \bigcup_{j:\ m_j \in G} \Omega_j\},$$

$$\mathbf{T}_3^* \triangleq \mathbf{T}^*(N_3, \eta).$$

Using Lemma H.11, we have

$$\limsup_{\eta\downarrow 0} \mathbb{P}_x\left(N_3 - N_2 > \frac{\Delta_t/4}{\lambda_G(\eta)} \,\Big|\, A_1\right) \leq \epsilon_1. \tag{H.54}$$

Meanwhile, on event $A_1 \cap \left\{N_3 - N_2 \leq \frac{\Delta_t/4}{\lambda_G(\eta)}\right\} \cap \left\{X_s^{*,\eta} \in \bigcup_{k:\, m_k \in G} \Omega_k \;\forall s \in [0,t]\right\}$, we have $\mathbf{T}_3^* - \mathbf{T}_2^* \leq \Delta_t/4$, hence $\mathbf{T}_3^* \in [t - \Delta_t, t - \Delta_t/4]$. In summary,

$$A_1 \cap \left\{N_3 - N_2 \leq \frac{\Delta_t/4}{\lambda_G(\eta)}\right\} \cap \left\{X_s^{*,\eta} \in \bigcup_{k:\, m_k \in G} \Omega_k \;\forall s \in [0,t]\right\}$$
$$\subseteq \left\{\mathbf{T}_3^* \in [t - \Delta_t, t - \Delta_t/4]\right\} \cap \left\{X_s^{*,\eta} \in \bigcup_{k:\, m_k \in G} \Omega_k \;\forall s \in [0, \mathbf{T}_3^*]\right\}$$

Moreover, on event $\left\{X_s^{*,\eta} \in \bigcup_{k:\, m_k \in G} \Omega_k \;\forall s \in [0,t]\right\}$, if we let $J_3$ be label of the local minimum visited at $\mathbf{T}_3^*$ such that $J_3 = j \iff X_{\mathbf{T}_3^*}^{*,\eta} \in B(m_j, 2\Delta)$, then we must have $m_{J_3} \in G^{\mathrm{large}}$. Meanwhile, consider the following stopping times

$$\mathbf{T}^\sigma \triangleq \min\{s > \mathbf{T}_3^* : X_s^{*,\eta} \notin \Omega_{J_3}\}.$$

From Proposition G.25, we know that

$$\limsup_{\eta\downarrow 0} \mathbb{P}_x\left(\mathbf{T}^\sigma - \mathbf{T}_3^* \leq \Delta_t \,\Big|\, \left\{\mathbf{T}_3^* \in [t - \Delta_t, t - \Delta_t/4]\right\} \cap \left\{X_s^{*,\eta} \in \bigcup_{k:\, m_k \in G} \Omega_k \;\forall s \in [0, \mathbf{T}_3^*]\right\}\right)$$
$$\leq \epsilon_1 + 1 - \exp\left(-(1 + \epsilon_1)q^*\Delta_t\right) \tag{H.55}$$

where $q^* = \max_j \mu_j(E_j)$. Now we define the event

$$A_2 \triangleq A_1 \cap \left\{N_3 - N_2 \leq \frac{\Delta_t/4}{\lambda_G(\eta)}\right\} \cap \left\{\mathbf{T}^\sigma - \mathbf{T}_3^* > \Delta_t\right\} \cap \left\{X_s^{*,\eta} \in \bigcup_{k:\, m_k \in G} \Omega_k \;\forall s \in [0, \mathbf{T}_3^*]\right\}.$$

Using eq. (H.53)-eq. (H.55), we get

$$\limsup_{\eta\downarrow 0} \sup_{x \in [-L,L]} \mathbb{P}_x\left(A_2^c \cap \{X_s^{*,\eta} \in \bigcup_{k:\, m_k \in G} \Omega_k \;\forall s \in [0,t]\}\right) \leq 2\epsilon_1 + 1 - \exp\left(-(1 + \epsilon_1)q^*\Delta_t\right). \tag{H.56}$$

Furthermore, on event $A_2$, due to $\mathbf{T}_3^* \in [t - \Delta_t, t - \Delta_t/4]$ as established above, we must have

$$X_s^{*,\eta} \in \Omega_{J_3} \;\forall s \in [\mathbf{T}_3^*, t].$$

Now let us focus on a timestamp $\mathbf{T}_4^* = t - \frac{\Delta_t/8}{H(1/\eta)}\lambda_G(\eta)$ and $N_4 = \mathbf{N}^*(\mathbf{T}_4^*, \eta)$. Obviously, $\mathbf{T}_4^* > \mathbf{T}_3^*$ on event $A_2$. Next, define

$$N_5 \triangleq \min\{n \geq N_4 : X_n^\eta \in \bigcup_j B(m_j, 2\Delta)\}$$
$$\mathbf{T}_5^* \triangleq \mathbf{T}^*(N_5, \eta).$$

Using Lemma H.7 and H.8 again as we did above when obtaining eq. (H.52), we can show that

$$\limsup_{\eta\downarrow 0} \mathbb{P}_x\left(N_5 - N_4 > \frac{\Delta_t/16}{H(1/\eta)}\right) = 0. \tag{H.57}$$

On the other hand, on event $A_2 \cap \left\{N_5 - N_4 \leq \frac{\Delta_t/16}{H(1/\eta)}\right\}$ we must have

- $\mathbf{T}_5^* - \mathbf{T}_4^* \leq \frac{\Delta_t/16}{H(1/\eta)}\lambda_G(\eta)$, so $\mathbf{T}_5^* \in [t - \frac{\Delta_t/8}{H(1/\eta)}\lambda_G(\eta), t - \frac{\Delta_t/16}{H(1/\eta)}\lambda_G(\eta)]$;

- $X_{\mathbf{T}_5^*}^{*,\eta} \in \Omega_{J_3}$, due to $\mathbf{T}^\sigma - \mathbf{T}_3^* > \Delta_t$.

This implies that for event

$$\widetilde{A} \triangleq \left\{ \mathbf{T}_5^* \in [t - \frac{\Delta_t/8}{H(1/\eta)}\lambda_G(\eta), t - \frac{\Delta_t/16}{H(1/\eta)}\lambda_G(\eta)], \ X_{\mathbf{T}_5^*}^{*,\eta} \in \Omega_{J_3} \right\} \cap \{X_s^{*,\eta} \in \bigcup_{k: \ m_k \in G} \Omega_j \ \forall s \in [0, \mathbf{T}_5^*]\},$$

we have $A_2 \cap \left\{ N_5 - N_4 \leq \frac{\Delta_t/16}{H(1/\eta)} \right\} \subseteq \widetilde{A}$. Lastly, observe that

- From Lemma G.4, we know that for $N_6(\delta) \triangleq \min\{n > N_5 : \eta|Z_n| > \delta\}$ we have

$$\limsup_{\eta \downarrow 0} \mathbb{P}\big(N_6(\delta) - N_5 \leq \Delta_t/H(1/\eta)\big) \leq \Delta_t/\delta^\alpha;$$

- As stated at the beginning of the proof, our choice of $\delta$ allows us to apply Lemma G.14 and show that

$$\limsup_{\eta \downarrow 0} \sup_{x \in [-L, L]} \mathbb{P}_x\big(\exists n = N_5, \cdots, N_6 - 1 \text{ s.t. } X_n^\eta \notin B(m_{J_3}, 3\Delta) \mid \widetilde{A}\big) = 0;$$

- Combining the two bullet points above, we get

$$\limsup_{\eta \downarrow 0} \mathbb{P}_x\Big(\exists s \in [\mathbf{T}_5^*, t] \text{ such that } X_s^{*,\eta} \notin B(m_{J_3}, 3\Delta) \ \Big| \ \widetilde{A}\Big) \leq \Delta_t/\delta^\alpha. \qquad \text{(H.58)}$$

On the other hand,

$$\widetilde{A} \cap \{X_s^{*,\eta} \in B(m_{J_3}, 3\Delta) \ \forall s \in [\mathbf{T}_5^*, t]\} \subseteq \{X_t^{*,\eta} \in \bigcup_{k: \ m_k \in G^{\text{large}}} B(m_k, 3\Delta)\}.$$

In summary, for event

$$A_3 \triangleq A_2 \cap \left\{ N_5 - N_4 \leq \frac{\Delta_t/16}{H(1/\eta)} \right\} \cap \left\{ X_s^{*,\eta} \in B(m_{J_3}, 3\Delta) \ \forall s \in [\mathbf{T}_5^*, t] \right\},$$

we have $A_3 \subseteq \{X_t^{*,\eta} \in \bigcup_{k: \ m_k \in G^{\text{large}}} B(m_k, 3\Delta)\}$. Besides, due to eq. (H.56)eq. (H.57)eq. (H.58), we get

$$\limsup_{\eta \downarrow 0} \sup_{x \in [-L, L]} \mathbb{P}_x\Big(A_3^c \cap \{X_s^{*,\eta} \in \bigcup_{k: \ m_k \in G} \Omega_k \ \forall s \in [0, t]\}\Big)$$

$$\leq 2\epsilon_1 + 1 - \exp\big(-(1 + \epsilon_1)q^*\Delta_t\big) + \frac{\Delta_t}{\delta^\alpha}.$$

Remember that $\delta, \epsilon_1, q^*$ are fixed constants while $\Delta_t$ can be made arbitrarily small, so by driving $\Delta_t$ to 0 we can conclude the proof. $\qquad \square$

Recall the definition of jump processes in Definition H.1. Central to the proof of Lemma H.4, the next result provides a set of sufficient conditions for the convergence of a sequence of such jump processes in the sense of finite dimensional distributions.

**Lemma H.13.** *For a sequence of processes $(Y^n)_{n \geq 1}$ that, for each $n \geq 1$, $Y^n$ is a $\big((U_j^n)_{j \geq 0}, (V_j^n)_{j \geq 0}\big)$ jump process, and a $\big((U_j)_{j \geq 0}, (V_j)_{j \geq 0}\big)$ jump process $Y$, if*

- $U_0 \equiv 0$;

- $(U_0^n, V_0^n, U_1^n, V_1^n, U_2^n, V_2^n, \cdots)$ *converges in distribution to* $(0, V_0, U_1, V_1, U_2, V_2, \cdots)$ *as $n \to \infty$;*

- *For any $x > 0$ and any $n \geq 1$,*

$$\mathbb{P}(U_1 + \cdots + U_n = x) = 0;$$

- *For any $x > 0$,*

$$\lim_{n \to \infty} \mathbb{P}(U_1 + U_2 + \cdots U_n > x) = 1,$$

*then the finite dimensional distribution of $Y^n$ converges to that of $Y$ in the following sense: for any $k \in \mathbb{N}$ and any $0 < t_1 < t_2 < \cdots < t_k < \infty$, the random element $(Y_{t_1}^n, \cdots, Y_{t_k}^n)$ converges in distribution to $(Y_{t_1}, \cdots, Y_{t_k})$ as $n \to \infty$.*

*Proof.* Fix some $k \in \mathbb{N}$ and $0 < t_1 < t_2 < \cdots < t_k < \infty$. For notational simplicity, let $t = t_k$. Let $(\mathbb{D}, \mathbf{d})$ be the metric space where $\mathbb{D} = \mathbb{D}_{[0,t]}$, the space of all càdlàg functions in $\mathbb{R}$ on the time interval $[0, t]$, and $\mathbf{d}$ is the Skorokhod metric defined as

$$\mathbf{d}(\zeta_1, \zeta_2) \triangleq \inf_{\lambda \in \Lambda} \|\zeta_1 - \zeta_2 \circ \lambda\| \vee \|\lambda - I\|$$

where $\Lambda$ is the set of all nondecreasing homeomorphism from $[0, t]$ onto itself, and $I(s) = s$ is the identity mapping. Also, we arbitrarily choose some $\epsilon \in (0, 1)$ and some open set $A \subseteq \mathbb{R}^k$.

From the assumption, we can find integer $J(\epsilon)$ such that $\mathbb{P}\left(\sum_{j=1}^{J(\epsilon)} U_j \le t\right) < \epsilon$, as well as an integer $N(\epsilon)$ such that, for all $n \ge N(\epsilon)$, we have $\mathbb{P}\left(\sum_{j=1}^{J(\epsilon)} U_j^n \le t\right) + \mathbb{P}\left(U_0^n \ge t_1\right) < \epsilon$. We fix such $J(\epsilon), N(\epsilon)$ (we may abuse the notation slightly and simply write $J, N$ when there is no ambiguity).

Using Skorokhod's representation theorem, we can construct a probability space $(\mathbf{\Omega}, \mathcal{F}, \mathbb{Q})$ that supports random variables $(\widetilde{U}_0^n, \widetilde{V}_0^n, \cdots, \widetilde{U}_J^n, \widetilde{V}_J^n)_{n \ge 1}$ and $(\widetilde{U}_0, \widetilde{V}_0, \cdots, \widetilde{U}_J, \widetilde{V}_J)$ and satisfies the following conditions:

- $\mathcal{L}(U_0^n, V_0^n, \cdots, U_J^n, V_J^n) = \mathcal{L}(\widetilde{U}_0^n, \widetilde{V}_0^n, \cdots, \widetilde{U}_J^n, \widetilde{V}_J^n)$ for all $n \ge 1$;

- $\mathcal{L}(U_0, V_0, \cdots, U_J, V_J) = \mathcal{L}(\widetilde{U}_0, \widetilde{V}_0, \cdots, \widetilde{U}_J, \widetilde{V}_J)$;

- $U_j^n \xrightarrow{a.s.} U_j$ and $V_j^n \xrightarrow{a.s.} V_j$ as $n \to \infty$ for all $j \in [J]$.

Therefore, on $(\mathbf{\Omega}, \mathcal{F}, \mathbb{Q})$ we can define the following random elements (taking values in the space of càdlàg functions):

$$Y_s^{n,\downarrow J} = \begin{cases} \widetilde{V}_0^n & \text{if } s < \widetilde{U}_0^n \\ \sum_{j=0}^J \widetilde{V}_j^n \mathbb{1}_{[\widetilde{U}_0^n + \widetilde{U}_1^n + \cdots + \widetilde{U}_j^n, \, \widetilde{U}_0^n + \widetilde{U}_1^n + \cdots + \widetilde{U}_{j+1}^n)}(s) & \text{otherwise} \end{cases},$$

$$Y_s^{\downarrow J} = \sum_{j=0}^J \widetilde{V}_j \mathbb{1}_{[\widetilde{U}_1 + \cdots + \widetilde{U}_j, \, \widetilde{U}_1 + \cdots + \widetilde{U}_{j+1})}(s) \quad \forall s \ge 0.$$

Note that (1) for the first jump time of $Y^{\downarrow J}$ we have $\widetilde{U}_0 \equiv 0$, hence $Y_0^{\downarrow J} = \widetilde{V}_0$; (2) when defining $Y^{n,\downarrow J}$ we set its value on $[0, \widetilde{U}_0^n)$ to be $\widetilde{V}_0^n$ instead of 0.

Since $U_j^n \xrightarrow{a.s.} U_j$ and $V_j^n \xrightarrow{a.s.} V_j$ as $n \to \infty$ for all $j \in [J]$, we must have

$$\lim_n \mathbf{d}(Y_s^{n,\downarrow J}, Y_s^{\downarrow J}) = 0$$

almost surely, which further implies that $Y_s^{n,\downarrow J} \Rightarrow Y_s^{\downarrow J}$ as $n \to \infty$ on $(\mathbb{D}, \mathbf{d})$. Now from our assumption that, for the jump times $U_1 + \cdots + U_j$, we have $\mathbb{P}(U_1 + \cdots + U_j = x) = 0 \; \forall x > 0, j \ge 1$, as well as (13.3) in Billingsley (2013), we then obtain

$$(Y_{t_1}^{n,\downarrow J}, \cdots, Y_{t_k}^{n,\downarrow J}) \Rightarrow (Y_{t_1}^{\downarrow J}, \cdots, Y_{t_k}^{\downarrow J}) \tag{H.59}$$

as $n \to \infty$. Recall that $A$ is the open set we arbitrarily chose at the beginning of the proof, and $\epsilon > 0$ is also chosen arbitrarily. Now we observe the following facts.

- Using eq. (H.59), we can see that

$$\liminf_n \mathbb{Q}\left((Y_{t_1}^{n,\downarrow J}, \cdots, Y_{t_k}^{n,\downarrow J}) \in A\right) \ge \mathbb{Q}\left(Y_{t_1}^{\downarrow J}, \cdots, Y_{t_k}^{\downarrow J}) \in A\right).$$

- The choice of $N(\epsilon)$ and $J(\epsilon)$ above implies that

$$\left| \mathbb{Q}\big((Y_{t_1}^{\downarrow J(\epsilon)}, \cdots, Y_{t_k}^{\downarrow J(\epsilon)}) \in A\big) - \mathbb{P}\big((Y_{t_1}, \cdots, Y_{t_k}) \in A\big)\right| \leq \mathbb{P}\Big(\sum_{j=1}^{J(\epsilon)} U_j \leq t\Big) < \epsilon,$$

$$\left| \mathbb{Q}\big((Y_{t_1}^{n, \downarrow J(\epsilon)}, \cdots, Y_{t_k}^{n, \downarrow J(\epsilon)}) \in A\big) - \mathbb{P}\big((Y_{t_1}^n, \cdots, Y_{t_k}^n) \in A\big)\right|$$

$$\leq \mathbb{P}\Big(\sum_{j=1}^{J(\epsilon)} U_j^n \leq t\Big) + \mathbb{P}\Big(U_0^n \geq t_1\Big) < \epsilon \ \forall n \geq N(\epsilon).$$

Collecting the two results above, we have established that

$$\liminf_n \mathbb{P}\big((Y_{t_1}^n, \cdots, Y_{t_k}^n) \in A\big) \geq \mathbb{P}\big((Y_{t_1}, \cdots, Y_{t_k}) \in A\big) - 2\epsilon.$$

From Portmanteau theorem, together with arbitrariness of $\epsilon > 0$ and open set $A$, we can now conclude that $(Y_{t_1}^n, \cdots, Y_{t_k}^n)$ converges in distribution to $(Y_{t_1}, \cdots, Y_{t_k})$. $\qquad \square$

The following lemma concerns the scaled version of the marker process $\hat{X}^{*,\eta,\Delta}$ defined in eq. (H.13)-eq. (H.14). Obviously, it is a jump process that complies with Definition H.1. When there is no ambiguity about the sequences $(\eta_n)_{n\geq 1}$ and $(\Delta_n)_{n\geq 1}$, let $\hat{X}_t^{(n)} \triangleq \hat{X}_t^{*,\eta_n,\Delta_n}$. From eq. (H.6)-eq. (H.11) and eq. (H.15), we know that for any $n \geq 1$, $\hat{X}^{(n)}$ is a $\Big(\big(\tau_k^*(\eta_n, \Delta_n) - \tau_{k-1}^*(\eta_n, \Delta_n)\big)_{k\geq 0}, \big(m_{I_k(\eta_n,\Delta_n)}\big)_{k\geq 0}\Big)$-jump process (with the convention that $\tau_{-1}^* = 0$). Also, for clarity of the exposition, we let (for all $n \geq 1, k \geq 0$)

$$\widetilde{S}_k^{(n)} = \sigma_k^*(\eta_n, \Delta_n) - \tau_{k-1}^*(\eta_n, \Delta_n),$$
$$S_k^{(n)} = \tau_k^*(\eta_n, \Delta_n) - \tau_{k-1}^*(\eta_n, \Delta_n),$$
$$\widetilde{W}_k^{(n)} = m_{\widetilde{I}_k^G(\eta_n, \Delta_n)},$$
$$W_k^{(n)} = m_{I_k^G(\eta_n, \Delta_n)}.$$

Lastly, remember that $Y$ is the continuous-time Markov chain defined in eq. (H.18)-eq. (H.21) and $\pi_G(\cdot)$ is the random mapping defined in eq. (H.22) that is used to initialize $Y$. Besides, $Y$ is a $\big((S_k)_{k\geq 0}, (W_k)_{k\geq 0}\big)$ jump process under Definition H.1, with $S_0 = 0$ and $W_0 = \pi_G(m_i)$ (here $x \in \Omega_i$ and $X_0^\eta = x$, so $i$ is the index of the attraction field where the SGD iterate is initialized). The following result states that, given a sequence of learning rates $(\eta_n)_{n\geq 1}$ that tend to 0, we are able to find a sequence of $(\Delta_n)_{n\geq 1}$ to parametrize $\hat{X}^{(n)} = \hat{X}^{*,\eta_n,\Delta_n}, X^{(n)} = X^{*,\eta_n,\Delta_n}$ so that they have several useful properties, one of which is that the jump times and locations of $\hat{X}^{(n)}$ converges in distribuiton to those of $Y(\pi_G(m_i))$.

**Lemma H.14.** *Assume the communication class $G$ is absorbing. Given any $m_i \in G$, $x \in \Omega_i$, finitely many real numbers $(t_l)_{l=1}^{k'}$ such that $0 < t_1 < t_2 < \cdots < t_{k'}$, and a sequence of strictly positive real numbers $(\eta_n)_{n\geq 1}$ with $\lim_{n\to 0} \eta_n = 0$, there exists a sequence of strictly positive real numbers $(\Delta_n)_{n\geq 1}$ with $\lim_n \Delta_n = 0$ such that*

- *Under $\mathbb{P}_x$ (so $X_0^\eta = x$), as $n$ tends to $\infty$,*

$$(S_0^{(n)}, W_0^{(n)}, S_1^{(n)}, W_1^{(n)}, S_2^{(n)}, W_2^{(n)}, \cdots) \Rightarrow (S_0, W_0, S_1, W_1, S_2, W_2, \cdots) \quad \text{(H.60)}$$

- *(Recall the definition of $T_k, I_k$ in eq. (H.39)-eq. (H.42)) Given any $\epsilon > 0$, the following claim holds for all $n$ sufficiently large:*

$$\sup_{k\geq 0} \mathbb{P}_x\Big(\exists j \in [T_k(\eta_n, \Delta_n), T_k(\eta_n, \Delta_n)] \ s.t. \ X_j^\eta \notin \bigcup_{l:\, m_l \in G} \Omega_l \mid m_{I_k(\eta_n, \Delta_n)} \in G\Big) < \epsilon;$$
$$\text{(H.61)}$$

- *Given any $\epsilon > 0$, the following claim holds for all $n$ sufficiently large,*

$$\sup_{k \geq 0} \mathbb{P}_k \Big( m_{I_k(\eta_n, \Delta_n) + v} \notin G^{large} \ \forall v \in [u n_{min}] \ \Big| \ m_{I_k(\eta_n, \Delta_n)} \in G \Big)$$
$$\leq \mathbb{P}(Geom(p^*) \geq u) + \epsilon \ \forall u = 1, 2, \cdots ; \qquad (\text{H.62})$$

- *For any $l \in [k']$,*

$$\lim_{n \to \infty} \mathbb{P}_x \Big( X^{*, \eta_n}_{t_l} \notin \bigcup_{j: \ m_j \in G^{large}} B(m_j, \Delta_n), \ X^{*, \eta_n}_s \in \bigcup_{j: \ m_j \in G} \Omega_j \ \forall s \in [0, t_{k'}] \Big) = 0$$
$$(\text{H.63})$$

*where $p^* > 0$ is a constant that does not vary with our choices of $\eta_n$ or $\Delta_n$.*

*Proof.* Let

$$\nu_j \triangleq q_j = \mu_j(E_j)$$
$$\nu_{j,k} \triangleq \mu_j(E_{j,k})$$

so from the definition of $q_{j,k}$ we have $q_{j,k} = \mathbb{1}\{j \neq k\} \nu_{j,k} + \sum_{l: \ m_l \in G^{small}} \nu_{j,l} p_{l,k}$.

In order to specify our choice of $(\Delta_n)_{n \geq 1}$, we consider a construction of sequences $(\bar{\boldsymbol{\Delta}}(j))_{j \geq 0}, (\bar{\boldsymbol{\eta}}(j))_{j \geq 0}$ as follows. Fix some $\theta \in (0, \alpha - 1)/2)$. Let $\bar{\boldsymbol{\Delta}}(0) = \bar{\boldsymbol{\eta}}(0) = 1$. One can see the existence of some $(\bar{\boldsymbol{\Delta}}(j))_{j \geq 1}, (\bar{\boldsymbol{\eta}}(j))_{j \geq 1}$ such that

- $\bar{\boldsymbol{\Delta}}(j) \in \big( 0, \bar{\boldsymbol{\Delta}}(j-1)/2 \big], \ \bar{\boldsymbol{\eta}}(j) \in \big( 0, \bar{\boldsymbol{\eta}}(j-1)/2 \big]$ for all $j \geq 1$;

- (Due to Lemma H.7) for any $j \geq 1$, $\eta \in (0, \bar{\boldsymbol{\eta}}(j)]$, (remember that $x$ and $i$ are the fixed constants prescribed in the description of the lemma)

$$\mathbb{P}_x \Big( \sigma_0^* \big( \eta, \bar{\boldsymbol{\Delta}}(j) \big) < \eta^\theta, \ \widetilde{I}_0^G \big( \eta, \bar{\boldsymbol{\Delta}}(j) \big) = i \Big) > 1 - \frac{1}{2^j}.$$

  For definitions of $\sigma_k^G, \tau_k^G, I_k^G, \widetilde{I}_k^G$, see eq. (H.6)-eq. (H.11).

- (Due to Lemma H.11) for any $j \geq 1$, $\eta \in (0, \bar{\boldsymbol{\eta}}(j)]$,

$$\left| \mathbb{P}_x \Big( \tau_k^* \big( \eta, \bar{\boldsymbol{\Delta}}(j) \big) - \sigma_k^* \big( \eta, \bar{\boldsymbol{\Delta}}(j) \big) < \eta^\theta, \ I_k^G \big( \eta, \bar{\boldsymbol{\Delta}}(j) \big) = i_2 \ \Big| \ \widetilde{I}_k^G \big( \eta, \bar{\boldsymbol{\Delta}}(j) \big) = i_1 \Big) - p_{i_1, i_2} \right|$$
$$< 1/2^j$$

  uniformly for all $k \geq 0$ and all $m_{i_1} \in G^{small}, m_{i_2} \in G^{large}$. Also, by definition of $\sigma^*$ and $\tau^*$, we must have

$$\mathbb{P}_x \Big( \tau_k^* \big( \eta, \bar{\boldsymbol{\Delta}}(j) \big) - \sigma_k^* \big( \eta, \bar{\boldsymbol{\Delta}}(j) \big) = 0, \ I_k^G \big( \eta, \bar{\boldsymbol{\Delta}}(j) \big) = i_1 \ \Big| \ \widetilde{I}_k^G \big( \eta, \bar{\boldsymbol{\Delta}}(j) \big) = i_1 \Big) = 1$$

  for all $k \geq 0$ and $m_{i_1} \in G^{large}$.

- (Due to Proposition G.25) for any $j \geq 1$, $\eta \in (0, \bar{\boldsymbol{\eta}}(j)]$,

$$- \frac{1}{2^j} + \exp \Big( - (1 + \frac{1}{2^j}) q_{i_1} u \Big) \frac{\nu_{i_1, i_2} - \frac{1}{2^j}}{q_{i_1}}$$
$$\leq \mathbb{P}_x \Big( \sigma_{k+1}^* \big( \eta, \bar{\boldsymbol{\Delta}}(j) \big) - \tau_k^* \big( \eta, \bar{\boldsymbol{\Delta}}(j) \big) > u, \ \widetilde{I}_{k+1}^G = i_2 \ \Big| \ I_k^G \big( \eta, \bar{\boldsymbol{\Delta}}(j) \big) = i_1 \Big)$$
$$\leq \frac{1}{2^j} + \exp \Big( - (1 - \frac{1}{2^j}) q_{i_1} u \Big) \frac{\nu_{i_1, i_2} + \frac{1}{2^j}}{q_{i_1}}$$

  uniformly for all $k \geq 1$, all $u > 1/2^j$, and all $m_{i_1} \in G^{large}, m_{i_2} \in G$.

- (Due to $G$ being absorbing and, again, Proposition G.25) for any $j \geq 1$, for any $j \geq 1$, $\eta \in (0, \bar{\boldsymbol{\eta}}(j)]$, (Recall the definition of $T_k, I_k$ in eq. (H.39)-eq. (H.42))

$$\mathbb{P}_x\Big(I_{k+1}\big(\eta, \bar{\boldsymbol{\Delta}}(j)\big) = i_2 \mid I_k\big(\eta, \bar{\boldsymbol{\Delta}}(j)\big) = i_1\Big) < \frac{1}{2^j} \tag{H.64}$$

  uniformly for all $k \geq 0$, $m_{i_1} \in G$, $m_{i_2} \notin G$.

- (Due to Lemma H.9) There exists some $p^* > 0$ such that for any $j \geq 1$, for any $j \geq 1$, $\eta \in (0, \bar{\boldsymbol{\eta}}(j)]$,

$$\mathbb{P}_x\Big(m_{v+I_k\big(\eta, \bar{\boldsymbol{\Delta}}(j)\big)} \notin G^{\text{large}} \;\forall v \in [un_{\min}] \;\Big|\; I_k\big(\eta, \bar{\boldsymbol{\Delta}}(j)\big) = i_1\Big)$$
$$\leq \mathbb{P}(Geom(p^*) \geq u) + 1/2^j \tag{H.65}$$

  uniformly for all $k \geq 0$, $u \geq 1$ and $m_{i_1} \in G$.

- (Due to Lemma H.12) for any $j \geq 1$, for any $j \geq 1$, $\eta \in (0, \bar{\boldsymbol{\eta}}(j)]$,

$$\mathbb{P}_x\Big(X^{*,\eta_n}_{t_k} \notin \bigcup_{j:\, m_j \in G^{\text{large}}} B(m_j, \bar{\boldsymbol{\Delta}}(j)),\; X^{*,\eta_n}_s \in \bigcup_{j:\, m_j \in G} \Omega_j \;\forall s \in [0, t_{k'}]\Big) < 1/2^j \tag{H.66}$$

  uniformly for all $k \in [k']$.

Fix such $(\bar{\boldsymbol{\Delta}}(j))_{j \geq 0}, (\bar{\boldsymbol{\eta}}(j))_{j \geq 0}$. Define a function $\mathbf{J}(\cdot) : \mathbb{N} \mapsto \mathbb{N}$ as

$$\mathbf{J}(n) = 0 \vee \max\{j \geq 0 : \bar{\boldsymbol{\eta}}(j) \geq \eta_n\}$$

with the convention that $\max \emptyset = -\infty$. Lastly, let

$$\Delta_n = \bar{\boldsymbol{\Delta}}(\mathbf{J}(n)) \;\forall n \geq 1.$$

Note that, due to $\lim_n \eta_n = 0$, we have $\lim_n \mathbf{J}(n) = \infty$, hence $\lim_n \Delta_n = 0$. Besides, the definition of $\mathbf{J}(\cdot)$ tells us that in case that $\mathbf{J}(n) \geq 1$ (which will hold for all $n$ sufficiently large), the claims above holds with $\eta = \eta_n$ and $j = \mathbf{J}(n)$. In particular, by combining $\lim_n \mathbf{J}(n) = \infty$ with eq. (H.64)eq. (H.65)eq. (H.66) respectively, we have eq. (H.61)eq. (H.62)eq. (H.63).

Now it remains to prove eq. (H.60). To this end, it suffices to show that, for any positive integer $K$, we have $(S_0^{(n)}, W_0^{(n)}, \cdots, S_K^{(n)}, W_K^{(n)})$ converges in distribution $(S_0, W_0, \cdots, S_K, W_K)$ as $n$ tends to infinity. In particular, note that $S_0 = 0, W_0 = \pi_G(m_i)$, so $W_0 = m_j$ with probability $p_{i,j}$ if $m_i \in G^{\text{small}}$, and $W_0 \equiv m_i$ if $m_i \in G^{\text{large}}$.

For clarity of the exposition, we restate some important claims above under the new notational system with $\widetilde{S}_k^{(n)}, \widetilde{W}_k^{(n)}, S_k^{(n)}, W_k^{(n)}$ we introduced right above this lemma. Given any $\epsilon > 0$, the following claims hold for all $n$ sufficiently large:

- First of all,

$$\mathbb{P}_x\Big(\widetilde{S}_0^{(n)} < \eta_n^\theta,\; \widetilde{W}_0^{(n)} = m_i\Big) > 1 - \epsilon. \tag{H.67}$$

- For all $k \geq 0$ and all $m_{i_1} \in G^{\text{small}}, m_{i_2} \in G^{\text{large}}$,

$$\left| \mathbb{P}_x\Big(S_k^{(n)} - \widetilde{S}_k^{(n)} < \eta_n^\theta,\; W_k^{(n)} = m_{i_2} \;\Big|\; \widetilde{W}_k^{(n)} = m_{i_1}\Big) - p_{i_1,i_2} \right| < \epsilon. \tag{H.68}$$

- For all $k \geq 0$ and all $m_{i_1} \in G^{\text{large}}$,

$$\mathbb{P}_x\Big(S_k^{(n)} - \widetilde{S}_k^{(n)} = 0,\; W_k^{(n)} = m_{i_1} \;\Big|\; \widetilde{W}_k^{(n)} = m_{i_1}\Big) = 1. \tag{H.69}$$

- For all $k \geq 0$, all $m_{i_1} \in G^{\text{large}}$, $m_{i_2} \in G$ and all $u > \epsilon$,

$$- \epsilon + \exp\left(-(1+\epsilon)q_{i_1}u\right)\frac{\nu_{i_1,i_2} - \epsilon}{q_{i_1}}$$

$$\leq \mathbb{P}_x\left(\widetilde{S}_{k+1}^{(n)} - S_k^{(n)} > u, \, \widetilde{W}_{k+1}^{(n)} = m_{i_2} \,\Big|\, W_k^{(n)} = m_{i_1}\right)$$

$$\leq \mathbb{P}_x\left(\widetilde{S}_{k+1}^{(n)} - S_k^{(n)} > u - \eta_n^\theta, \, \widetilde{W}_{k+1}^{(n)} = m_{i_2} \,\Big|\, W_k^{(n)} = m_{i_1}\right)$$

$$\leq \epsilon + \exp\left(-(1-\epsilon)q_{i_1}u\right)\frac{\nu_{i_1,i_2} + \epsilon}{q_{i_1}} \tag{H.70}$$

- Here is one implication of eq. (H.68). Since $|G| \leq n_{\min}$, we have

$$\mathbb{P}_x\left(S_k^{(n)} - \widetilde{S}_k^{(n)} \geq \eta_n^\theta \,\Big|\, \widetilde{W}_k^{(n)} = m_{i_1}\right) < n_{\min} \cdot \epsilon \tag{H.71}$$

for all $k \geq 0$ and $m_{i_1} \in G^{\text{small}}$.

- Note that for any $m_{i_1}, m_{i_2} \in G^{\text{large}}$ and any $k \geq 0$

$$\mathbb{P}_x\left(S_{k+1}^{(n)} - S_k^{(n)} > u, \, W_{k+1}^{(n)} = m_{i_2} \,\Big|\, W_k^{(n)} = m_{i_1}\right)$$

$$= \mathbb{1}\{i_2 \neq i_1\}\mathbb{P}_x\left(\widetilde{S}_{k+1}^{(n)} - S_k^{(n)} > u, \, \widetilde{W}_{k+1}^{(n)} = m_{i_2} \,\Big|\, W_k^{(n)} = m_{i_1}\right)$$

$$+ \sum_{i_3:\, m_{i_3} \in G^{\text{small}}} \int_{s>0} \mathbb{P}_x\left(S_{k+1}^{(n)} - \widetilde{S}_{k+1}^{(n)} \geq (u-s) \vee 0, \, \widetilde{W}_{k+1}^{(n)} = m_{i_2} \,\Big|\, \widetilde{W}_k^{(n)} = m_{i_3}\right)$$

$$\cdot \mathbb{P}_x\left(\widetilde{S}_{k+1}^{(n)} - S_k^{(n)} = ds, \, \widetilde{W}_{k+1}^{(n)} = m_{i_3} \,\Big|\, W_k^{(n)} = m_{i_1}\right).$$

Fix some $i_3$ with $m_{i_3} \in G^{\text{small}}$. On the one hand, due to eq. (H.71),

$$\int_{s \in (0, u - \eta_n^\theta]} \mathbb{P}_x\left(S_{k+1}^{(n)} - \widetilde{S}_{k+1}^{(n)} \geq u - s, \, \widetilde{W}_{k+1}^{(n)} = m_{i_2} \,\Big|\, \widetilde{W}_k^{(n)} = m_{i_3}\right)$$

$$\cdot \mathbb{P}_x\left(\widetilde{S}_{k+1}^{(n)} - S_k^{(n)} = ds, \, \widetilde{W}_{k+1}^{(n)} = m_{i_3} \,\Big|\, W_k^{(n)} = m_{i_1}\right)$$

$$\leq n_{\min}\epsilon.$$

On the other hand, by considering the integral on $(u - \eta_n^\theta, \infty)$, we get

$$\int_{s \in (u - \eta_n^\theta, \infty)} \mathbb{P}_x\left(S_{k+1}^{(n)} - \widetilde{S}_{k+1}^{(n)} \geq (u-s) \vee 0, \, \widetilde{W}_{k+1}^{(n)} = m_{i_2} \,\Big|\, \widetilde{W}_k^{(n)} = m_{i_3}\right)$$

$$\cdot \mathbb{P}_x\left(\widetilde{S}_{k+1}^{(n)} - S_k^{(n)} = ds, \, \widetilde{W}_{k+1}^{(n)} = m_{i_3} \,\Big|\, W_k^{(n)} = m_{i_1}\right)$$

$$\geq \int_{s \in (u, \infty)} \mathbb{P}_x\left(\widetilde{W}_{k+1}^{(n)} = m_{i_2} \,\Big|\, \widetilde{W}_k^{(n)} = m_{i_3}\right)$$

$$\cdot \mathbb{P}_x\left(\widetilde{S}_{k+1}^{(n)} - S_k^{(n)} = ds, \, \widetilde{W}_{k+1}^{(n)} = m_{i_3} \,\Big|\, W_k^{(n)} = m_{i_1}\right)$$

$$\geq (p_{i_3, i_2} - \epsilon)\left(-\epsilon + \exp\left(-(1+\epsilon)q_{i_1}u\right)\frac{\nu_{i_1, i_3} - \epsilon}{q_{i_1}}\right)$$

due to eq. (H.68) and eq. (H.70). Meanwhile,

$$\int_{s \in (u - \eta_n^\theta, \infty)} \mathbb{P}_x\left(S_{k+1}^{(n)} - \widetilde{S}_{k+1}^{(n)} \geq (u-s) \vee 0, \, \widetilde{W}_{k+1}^{(n)} = m_{i_2} \,\Big|\, \widetilde{W}_k^{(n)} = m_{i_3}\right)$$

$$\cdot \mathbb{P}_x\left(\widetilde{S}_{k+1}^{(n)} - S_k^{(n)} = ds, \, \widetilde{W}_{k+1}^{(n)} = m_{i_3} \,\Big|\, W_k^{(n)} = m_{i_1}\right)$$

$$\leq (n_{\min}\epsilon + p_{i_3, i_2} + \epsilon)\left(\epsilon + \exp\left(-(1-\epsilon)q_{i_1}u\right)\frac{\nu_{i_1, i_3} + \epsilon}{q_{i_1}}\right)$$

due to eq. (H.68), eq. (H.70) and eq. (H.71).

- Therefore, for any $m_{i_1}, m_{i_2} \in G^{\text{large}}$ and any $k \geq 0$,

$$\mathbb{P}_x\left(S_{k+1}^{(n)} - S_k^{(n)} > u, \; W_{k+1}^{(n)} = m_{i_2} \;\Big|\; W_k^{(n)} = m_{i_1}\right)$$

$$\leq g(\epsilon) + \exp\left(-(1-\epsilon)q_{i_1}u\right)\frac{\mathbb{1}\{i_2 \neq i_1\}\nu_{i_1,i_2} + \sum_{i_3:\, m_{i_3} \in G^{\text{small}}} \nu_{i_1,i_3}p_{i_3,i_2}}{q_{i_1}}$$

$$\leq g(\epsilon) + \exp\left(-(1-\epsilon)q_{i_1}u\right)\frac{q_{i_1,i_2}}{q_{i_1}} \tag{H.72}$$

and

$$\mathbb{P}_x\left(S_{k+1}^{(n)} - S_k^{(n)} > u, \; W_{k+1}^{(n)} = m_{i_2} \;\Big|\; W_k^{(n)} = m_{i_1}\right)$$

$$\geq -g(\epsilon) + \exp\left(-(1+\epsilon)q_{i_1}u\right)\frac{\mathbb{1}\{i_2 \neq i_1\}\nu_{i_1,i_2} + \sum_{i_3:\, m_{i_3} \in G^{\text{small}}} \nu_{i_1,i_3}p_{i_3,i_2}}{q_{i_1}}$$

$$\geq -g(\epsilon) + \exp\left(-(1+\epsilon)q_{i_1}u\right)\frac{q_{i_1,i_2}}{q_{i_1}} \tag{H.73}$$

where $q^* = \max_i q_i$ and

$$g(\epsilon) \triangleq \epsilon + \frac{\epsilon}{q^*} + n_{\min}(1+\epsilon)\epsilon + \epsilon\frac{1+\epsilon}{q^*}n_{\min}$$

$$+ n_{\min}(\epsilon + \frac{\epsilon}{q^*}) + n_{\min}(n_{\min}\epsilon + \epsilon + 1)(1 + \frac{1}{q^*})\epsilon.$$

Note that $\lim_{\epsilon \downarrow 0} g(\epsilon) = 0$.

Now we apply the bounds in eq. (H.67)eq. (H.72)eq. (H.73) to establish the weak convergence claim regarding $(S_0^{(n)}, W_0^{(n)}, \cdots, S_K^{(n)}, W_K^{(n)})$. Fix some positive integer $K$, some strictly positive real numbers $(s_k)_{k=0}^K$, a sequence $(w_k)_{k=0}^K \in \left(G^{\text{large}}\right)^{K+1}$ with $w_k = m_{i_k}$ for each $k$, and some $\epsilon > 0$ such that $\epsilon < \min_{k=0,1,\cdots,K}\{s_k\}$. On the one hand, the definition of the CTMC $Y$ implies that

$$\mathbb{P}\left(S_0 < t_0, W_0 = w_0; \; S_k > s_k \text{ and } W_k = w_k \; \forall k \in [K]\right)$$

$$=\mathbb{P}(\pi_G(m_i) = w_0)\prod_{k=1}^K \left(S_k > s_k \text{ and } W_k = w_k \;\Big|\; W_{k-1} = w_{k-1}\right)$$

$$=\left(\mathbb{1}\{m_i \in G^{\text{large}}, \; i_0 = i\} + \mathbb{1}\{m_i \in G^{\text{small}}\}p_{i_0,i_1}\right) \cdot \prod_{k=1}^K \exp(-q_{i_{k-1}}s_k)\frac{q_{i_{k-1},i_k}}{q_{i_{k-1}}}.$$

On the other hand, using eq. (H.67)eq. (H.72)eq. (H.73), we know that for all $n$ sufficiently large,

$$\mathbb{P}_x\left(S_0^{(n)} < s_0, W_0^{(n)} = w_0; \; S_k^{(n)} > s_k \text{ and } W_k^{(n)} = w_k \; \forall k \in [K]\right)$$

$$\geq (1-\epsilon)\left(\mathbb{1}\{m_i \in G^{\text{large}}, \; i_0 = i\} + \mathbb{1}\{m_i \in G^{\text{small}}\}(p_{i_0,i_1} - \epsilon)\right)$$

$$\cdot \prod_{k=1}^K \left(-g(\epsilon) + \exp(-(1+\epsilon)q_{i_{k-1}}s_k)\frac{q_{i_{k-1},i_k}}{q_{i_{k-1}}}\right)$$

and

$$\mathbb{P}_x\left(S_0^{(n)} < s_0, W_0^{(n)} = w_0; \; S_k^{(n)} > s_k \text{ and } W_k^{(n)} = w_k \; \forall k \in [K]\right)$$

$$\leq \left(\mathbb{1}\{m_i \in G^{\text{large}}, \; i_0 = i\} + \mathbb{1}\{m_i \in G^{\text{small}}\}(p_{i_0,i_1} + \epsilon)\right)$$

$$\cdot \prod_{k=1}^K \left(g(\epsilon) + \exp(-(1-\epsilon)q_{i_{k-1}}s_k)\frac{q_{i_{k-1},i_k}}{q_{i_{k-1}}}\right).$$

Since $\epsilon > 0$ can be arbitrarily small, we now obtain

$$\lim_{n\to\infty} \mathbb{P}_x \Big( S_0^{(n)} < t_0, W_0^{(n)} = w_0;\ S_k^{(n)} > s_k \text{ and } W_k^{(n)} = m_k\ \forall k \in [K] \Big)$$

$$= \mathbb{P}\Big( S_0 < s_0, W_0 = w_0;\ S_k > s_k \text{ and } W_k = w_k\ \forall k \in [K] \Big),$$

and the arbitrariness of the integer $K$, the strictly positive real numbers $(s_k)_{k=0}^K$, and the sequence $(w_k)_{k=0}^K \in \big(G^{\text{large}}\big)^{K+1}$ allows us to conclude the proof. $\qquad\square$

To extend the result above to the case where the communication class $G$ is transient, we revisit the definition of the $Y^\dagger$ in eq. (H.26). Let $\bar{G} = G^{\text{large}} \cup \{\dagger\}$ and let $m_0 = \dagger$ (remember that all the local minimizers of $f$ on $[-L, L]$ are $m_1, \cdots, m_{\min}$). Meanwhile, using $q_i$ and $q_{i,j}$ in eq. (H.20)eq. (H.21), we can define

$$q_{i,j}^\dagger = \begin{cases} q_{i,j} & \text{if } i \geq 1,\ j \geq 1 \\ \mathbb{1}\{j = 0\} & \text{if } i = 0 \\ \sum_{j \in [n_{\min}], m_j \notin G} q_{i,j} & \text{if } i \geq 1,\ j = 0. \end{cases}$$

and $q_0^\dagger = 1$, $q_i^\dagger = q_i\ \forall i \geq 1$. Next, fix some $i$ with $m_i \in G$ and $x \in \Omega_i$. Define a sequence of random variables $(S_k^\dagger)_{k\geq 0}, (W_k^\dagger)_{k\geq 0}$ such that $S_k^\dagger = 0$ and $W_0 = \pi_G(m_i), W_0^\dagger = \dagger\mathbb{1}\{W_0 \notin G^{\text{large}}\} + W_0\mathbb{1}\{W_0 \in G^{\text{large}}\}$ (see the definition of random mapping $\pi_G$ in eq. (H.22)) and (for all $k \geq 0$ and $i, j$ with $m_j, m_l \in \bar{G}$)

$$\mathbb{P}\Big( W_{k+1}^\dagger = m_l,\ S_{k+1}^\dagger > t \,\Big|\, W_k^\dagger = m_j,\ (W_l^\dagger)_{l=0}^{k-1},\ (S_l^\dagger)_{l=0}^k \Big) \tag{H.74}$$

$$= \mathbb{P}\Big( W_{k+1}^\dagger = m_l,\ S_{k+1}^\dagger > t \,\Big|\, W_k^\dagger = m_j \Big) = \exp(-q_j^\dagger t)\frac{q_{j,l}^\dagger}{q_j^\dagger}\ \forall t > 0 \tag{H.75}$$

Then it is easy to see that $Y^\dagger(\pi_G(m_i))$ defined in eq. (H.26) is a $\big((S_k^\dagger)_{k\geq 0}, (W_k^\dagger)_{k\geq 0}\big)$ jump process. In particular, from at any state that is not $\dagger$ (namely, any $m_j$ with $m_j \in G^{\text{large}}$), the probability that $Y^\dagger$ moves to $\dagger$ in the next transition is equal to the chance that, starting from the same state, $Y$ moves to a state that is not in $G$. Once entering $m_0 = \dagger$, the process $Y^\dagger$ will only make dummy jumps (with interarrival times being iid Exp(1)): indeed, we have $q_0^\dagger = q_{0,0}^\dagger = 1$ and $q_{0,j}^\dagger = 0$ for any $j \geq 1$, implying that, given $W_k^\dagger = m_0 = \dagger$, we must have $W_{k+1}^\dagger = m_0 = \dagger$. These dummy jumps ensure that $Y^\dagger$ is stuck at the cemetery state $\dagger$ after visiting it.

Similarly, we can characterize the jump times and locations of the jump process $\hat{X}^{\dagger,*,\eta,\Delta}$ (for the definition, see eq. (H.25)). When there is no ambiguity about the sequences $(\eta_n)_{n\geq 1}, (\Delta_n)_{n\geq 1}$, let $\hat{X}^{\dagger,(n)} = \hat{X}^{*,\eta_n,\Delta_n}$ and $X^{\dagger,(n)} = X^{*,\eta_n,\Delta_n}$. Also, recall that $\tau_G$ defined in eq. (H.4) is the step $n$ when $X_n^\eta$ exits the communication class $G$. Now let $(E_k)_{k\geq 0}$ be a sequence of iid Exp(1) random variables that is also independent of the noises $(Z_k)_{k\geq 1}$ (so they are independent from the SGD iterates $X_n^\eta$). For all $n \geq 1, k \geq 0$, define (see eq. (H.6)-eq. (H.11) and eq. (H.15) for definitions of the quantities involved)

$$\widetilde{S}_k^{\dagger,(n)} = \begin{cases} \sigma_k^*(\eta_n, \Delta_n) \wedge \mathbf{T}^*(\tau_G(\eta_n), \eta_n) - \tau_{k-1}^*(\eta_n, \Delta_n) & \text{if } \tau_{k-1}^*(\eta_n, \Delta_n) < \mathbf{T}^*(\tau_G(\eta_n), \eta_n) \\ 0 & \text{otherwise} \end{cases}$$

$$S_k^{\dagger,(n)} = \begin{cases} \tau_k^*(\eta_n, \Delta_n) \wedge \mathbf{T}^*(\tau_G(\eta_n), \eta_n) - \tau_{k-1}^*(\eta_n, \Delta_n) & \text{if } \tau_{k-1}^*(\eta_n, \Delta_n) < \mathbf{T}^*(\tau_G(\eta_n), \eta_n) \\ E_k & \text{otherwise} \end{cases}$$

$$\widetilde{W}_k^{\dagger,(n)} = \begin{cases} m_{\widetilde{I}_k^G(\eta_n, \Delta_n)} & \text{if } \sigma_k^*(\eta_n, \Delta_n) < \mathbf{T}^*(\tau_G(\eta_n), \eta_n) \\ \dagger & \text{otherwise} \end{cases}$$

$$W_k^{\dagger,(n)} = \begin{cases} m_{I_k^G(\eta_n, \Delta_n)} & \text{if } \tau_k^*(\eta_n, \Delta_n) < \mathbf{T}^*(\tau_G(\eta_n), \eta_n) \\ \dagger & \text{otherwise} \end{cases}$$

with the convention that $\tau_{-1}^* = 0$. Note that $\mathbf{T}^*(\tau_G(\eta_n), \eta_n)$ is the scaled timestamp for $X^{(n)} = X^{*,\eta}$ corresponding to $\tau_G(\eta_n)$, hence $\mathbf{T}^*(\tau_G(\eta_n), \eta_n) = \min\{t \geq 0 : X^{(n)} \notin \bigcup_{j:\, m_j \in G} \Omega_j\}$. One can

see that $\hat{X}^{\dagger,(n)}$ is a $\big((S_k^{\dagger,(n)})_{k\geq 0}, (W_k^{\dagger,(n)})_{k\geq 0}\big)$ jump process. The next lemma is similar to Lemma H.14 and discusses the convergence of the jump times and locations of $\hat{X}^{\dagger,(n)}$ on a communication class $G$ in the transient case.

**Lemma H.15.** *Assume that the communication class $G$ is transient. Given any $m_i \in G$, $x \in \Omega_i$, finitely many real numbers $(t_l)_{l=1}^{k'}$ such that $0 < t_1 < t_2 < \cdots < t_{k'}$, and a sequence of strictly positive real numbers $(\eta_n)_{n\geq 1}$ with $\lim_{n\to 0}\eta_n = 0$, there exists a sequence of strictly positive real numbers $(\Delta_n)_{n\geq 1}$ with $\lim_n \Delta_n = 0$ such that*

- *Under $\mathbb{P}_x$ (so $X_0^\eta = x$), as $n$ tends to $\infty$,*

$$(S_0^{\dagger,(n)}, W_0^{\dagger,(n)}, S_1^{\dagger,(n)}, W_1^{\dagger,(n)}, S_2^{\dagger,(n)}, W_2^{\dagger,(n)}, \cdots) \Rightarrow (S_0^\dagger, W_0^\dagger, S_1^\dagger, W_1^\dagger, S_2^\dagger, W_2^\dagger, \cdots)$$
(H.76)

- *For any $l \in [k']$,*

$$\lim_{n\to\infty} \mathbb{P}_x\Big(X_{t_l}^{\dagger,(n)} \notin \bigcup_{j:\, m_j \in G^{large}} B(m_j, \Delta_n),\ X_s^{\dagger,(n)} \in \bigcup_{j:\, m_j \in G} \Omega_j\ \forall s \in [0, t_l]\Big) = 0$$
(H.77)

*Proof.* Let

$$\nu_{j,k} \triangleq \mu_j(E_{j,k})\ \forall j, k \geq 1,\ j \neq k$$

$$p_{j,\dagger} \triangleq \sum_{\widetilde{j}:\, m_{\widetilde{j}} \notin G} p_{j,\widetilde{j}}\quad \forall m_j \in G^{small}$$

$$q_{j,\dagger} \triangleq \sum_{k:\, m_k \notin G} \nu_{j,k} + \sum_{k:\, m_k \in G^{small}} \nu_{j,k} p_{k,\dagger}\quad \forall m_j \in G^{large}.$$

In order to specify our choice of $(\Delta_n)_{n\geq 1}$, we consider a construction of sequences $(\bar{\Delta}(j))_{j\geq 0}, (\bar{\eta}(j))_{j\geq 0}$ as follows. Fix some $\theta \in (0, \alpha-1)/2)$. Let $\bar{\Delta}(0) = \bar{\eta}(0) = 1$. One can see the existence of some $(\bar{\Delta}(j))_{j\geq 1}, (\bar{\eta}(j))_{j\geq 1}$ such that

- $\bar{\Delta}(j) \in \big(0, \bar{\Delta}(j-1)/2\big],\ \bar{\eta}(j) \in \big(0, \bar{\eta}(j-1)/2\big]$ for all $j \geq 1$;

- (Due to Lemma H.7) for any $j \geq 1$, $\eta \in (0, \bar{\eta}(j)]$, (remember that $x$ and $i$ are the fixed constants prescribed in the description of the lemma)

$$\mathbb{P}_x\Big(\sigma_0^*\big(\eta, \bar{\Delta}(j)\big) < \eta^\theta,\ \widetilde{I}_0^G\big(\eta, \bar{\Delta}(j)\big) = i\Big) > 1 - \frac{1}{2^j}.$$

For definitions of $\sigma_k^G, \tau_k^G, I_k^G, \widetilde{I}_k^G$, see eq. (H.6)-eq. (H.11).

- (Due to Lemma H.11) for any $j \geq 1$, $\eta \in (0, \bar{\eta}(j)]$,

$$\Big|\mathbb{P}_x\Big(\tau_k^*\big(\eta, \bar{\Delta}(j)\big) - \sigma_k^*\big(\eta, \bar{\Delta}(j)\big) < \eta^\theta,\ I_k^G\big(\eta, \bar{\Delta}(j)\big) = i_2\ \Big|\ \widetilde{I}_k^G\big(\eta, \bar{\Delta}(j)\big) = i_1\Big) - p_{i_1,i_2}\Big|$$

$$< 1/2^j$$

uniformly for all $k \geq 0$ and all $m_{i_1} \in G^{small}, m_{i_2} \notin G^{small}$. Also, by definition of $\sigma^*$ and $\tau^*$, we must have

$$\mathbb{P}_x\Big(\tau_k^*\big(\eta, \bar{\Delta}(j)\big) - \sigma_k^*\big(\eta, \bar{\Delta}(j)\big) = 0,\ I_k^G\big(\eta, \bar{\Delta}(j)\big) = i_1\ \Big|\ \widetilde{I}_k^G\big(\eta, \bar{\Delta}(j)\big) = i_1\Big) = 1$$

for all $k \geq 0$ and $m_{i_1} \in G^{large}$.

- (Due to Proposition G.25) for any $j \geq 1$, $\eta \in (0, \bar{\eta}(j)]$,

$$-\frac{1}{2^j} + \exp\Big(-(1 + \frac{1}{2^j})q_{i_1}u\Big)\frac{\nu_{i_1,i_2} - \frac{1}{2^j}}{q_{i_1}}$$

$$\leq \mathbb{P}_x\Big(\sigma_{k+1}^*\big(\eta, \bar{\Delta}(j)\big) - \tau_k^*\big(\eta, \bar{\Delta}(j)\big) > u,\ \widetilde{I}_{k+1}^G = i_2\ \Big|\ I_k^G\big(\eta, \bar{\Delta}(j)\big) = i_1\Big)$$

$$\leq \frac{1}{2^j} + \exp\Big(-(1 - \frac{1}{2^j})q_{i_1}u\Big)\frac{\nu_{i_1,i_2} + \frac{1}{2^j}}{q_{i_1}}$$

uniformly for all $k \geq 1$, all $u > 1/2^j$, and all $m_{i_1} \in G^{\text{large}}, m_{i_2} \in G$.

- (Due to Lemma H.12) for any $j \geq 1$, for any $j \geq 1$, $\eta \in (0, \bar{\boldsymbol{\eta}}(j)]$,

$$\mathbb{P}_x\Big(X_{t_k}^{(n)} \notin \bigcup_{j:\, m_j \in G^{\text{large}}} B(m_j, \bar{\boldsymbol{\Delta}}(j)),\ X_s^{(n)} \in \bigcup_{j:\, m_j \in G} \Omega_j\ \forall s \in [0, t_k]\Big) < 1/2^j \quad \text{(H.78)}$$

  uniformly for all $k \in [k']$.

Fix such $(\bar{\boldsymbol{\Delta}}(j))_{j \geq 0}, (\bar{\boldsymbol{\eta}}(j))_{j \geq 0}$. Define a function $\mathbf{J}(\cdot) : \mathbb{N} \mapsto \mathbb{N}$ as

$$\mathbf{J}(n) = 0 \vee \max\{j \geq 0 : \bar{\boldsymbol{\eta}}(j) \geq \eta_n\}$$

with the convention that $\max \emptyset = -\infty$. Lastly, let

$$\Delta_n = \bar{\boldsymbol{\Delta}}(\mathbf{J}(n)) \ \forall n \geq 1.$$

Note that, due to $\lim_n \eta_n = 0$, we have $\lim_n \mathbf{J}(n) = \infty$, hence $\lim_n \Delta_n = 0$. Besides, since $X_t^{\dagger,(n)} = X_t^{(n)}$ given $X_s^{(n)} \in \bigcup_{j:\, m_j \in G} \Omega_j$ for all $s \in [0, t]$, by combining $\lim_n \mathbf{J}(n) = \infty$ with eq. (H.78) we obtain eq. (H.77).

Now it remains to prove eq. (H.76). To this end, it suffices to show that, for any positive integer $K$, we have $(S_0^{\dagger,(n)}, W_0^{\dagger,(n)}, \cdots, S_K^{\dagger,(n)}, W_K^{\dagger,(n)})$ converges in distribution $(S_0^\dagger, W_0^\dagger, \cdots, S_K^\dagger, W_K^\dagger)$ as $n$ tends to infinity. In particular, due to introduction of the dummy jumps, we know that for any $k$ with $\tau_k^*(\eta_n, \Delta_n) \geq \tau_G(\eta_n)$ (in other words, $\hat{X}^{\dagger,(n)}$ has reached state $\dagger$ within the first $k$ jumps) we have $S_{k+1}^{\dagger,(n)} \sim \text{Exp}(1)$ and $W_{k+1}^{\dagger,(n)} \equiv \dagger$. Similarly, for any $k$ with $S_0^\dagger + \cdots + S_k^\dagger \leq \tau_G^Y$, we have $S_{k+1}^\dagger \sim \text{Exp}(1)$ and $W_{k+1}^\dagger \equiv \dagger$. Therefore, it suffices to show that, for any positive integer $K$, any series of strictly positive real numbers $(s_k)_{k=0}^K$, any sequence $(w_k)_{k=0}^K \in (\bar{G})^{K+1}$ such that $w_j \neq \dagger$ for any $j < K$, indices $i_k$ such that $w_k = m_{i_k}$ for each $k$, we have

$$\lim_{n \to \infty} \mathbb{P}_x\Big(S_0^{\dagger,(n)} < t_0, W_0^{\dagger,(n)} = w_0;\ S_k^{\dagger,(n)} > s_k \text{ and } W_k^{\dagger,(n)} = w_k\ \forall k \in [K]\Big)$$

$$= \mathbb{P}\Big(S_0^\dagger < s_0, W_0^\dagger = w_0;\ S_k^\dagger > s_k \text{ and } W_k^\dagger = w_k\ \forall k \in [K]\Big) \quad \text{(H.79)}$$

Fix some $(s_k)_{k=0}^K, (w_k)_{k=0}^K \in (\bar{G})^{K+1}$, and indices $(i_k)_{k=1}^K$ satisfying the conditions above. Besides, arbitrarily choose some $\epsilon > 0$ so that $\epsilon < \min_{k=0,\cdots,K} s_k$. To proceed, we start by translating the inequalities established above under the new system of notations.

- First of all, for all $n$ sufficiently large, (remember that $x$ and $i$ are prescribed constants in the description of this lemma)

$$\mathbb{P}_x\Big(\widetilde{S}_0^{\dagger,(n)} < \eta_n^\theta,\ \widetilde{W}_0^{\dagger,(n)} = m_i\Big) > 1 - \epsilon. \quad \text{(H.80)}$$

- For all $k \geq 0$ and all $m_{i_1} \in G^{\text{small}}, m_{i_2} \in G^{\text{large}}$, it holds for all $n$ sufficiently large that

$$\left| \mathbb{P}_x\Big(S_k^{\dagger,(n)} - \widetilde{S}_k^{\dagger,(n)} < \eta_n^\theta,\ W_k^{\dagger,(n)} = m_{i_2}\ \Big|\ \widetilde{W}_k^{\dagger,(n)} = m_{i_1}\Big) - p_{i_1, i_2} \right| < \epsilon. \quad \text{(H.81)}$$

- On the other hand, for all $k \geq 0$ and all $m_{i_1} \in G^{\text{small}}$, it holds for all $n$ sufficiently large that

$$\left| \mathbb{P}_x\Big(S_k^{\dagger,(n)} - \widetilde{S}_k^{\dagger,(n)} < \eta_n^\theta,\ W_k^{\dagger,(n)} = \dagger\ \Big|\ \widetilde{W}_k^{\dagger,(n)} = m_{i_1}\Big) - \sum_{i_2:\, m_{i_2} \notin G} p_{i_1, i_2} \right|$$

$$= \left| \mathbb{P}_x\Big(S_k^{\dagger,(n)} - \widetilde{S}_k^{\dagger,(n)} < \eta_n^\theta,\ W_k^{\dagger,(n)} = \dagger\ \Big|\ \widetilde{W}_k^{\dagger,(n)} = m_{i_1}\Big) - p_{i_1, \dagger} \right| < \epsilon. \quad \text{(H.82)}$$

- For all $k \geq 0$ and all $m_{i_1} \in G^{\text{large}}$, it holds for all $n$ that

$$\mathbb{P}_x\Big(S_k^{\dagger,(n)} - \widetilde{S}_k^{\dagger,(n)} = 0,\ W_k^{\dagger,(n)} = m_{i_1}\ \Big|\ \widetilde{W}_k^{\dagger,(n)} = m_{i_1}\Big) = 1. \quad \text{(H.83)}$$

- For all $k \geq 0$, all $m_{i_1} \in G^{\text{large}}$, $m_{i_2} \in G$ and all $u > \epsilon$, the following claim holds for all $n$ sufficiently large:

$$- \epsilon + \exp\left(- (1+\epsilon)q_{i_1}u\right)\frac{\nu_{i_1,i_2} - \epsilon}{q_{i_1}}$$
$$\leq \mathbb{P}_x\left(\widetilde{S}_{k+1}^{\dagger,(n)} - S_k^{\dagger,(n)} > u, \ \widetilde{W}_{k+1}^{\dagger,(n)} = m_{i_2} \ \Big| \ W_k^{\dagger,(n)} = m_{i_1}\right)$$
$$\leq \mathbb{P}_x\left(\widetilde{S}_{k+1}^{\dagger,(n)} - S_k^{\dagger,(n)} > u - \eta_n^\theta, \ \widetilde{W}_{k+1}^{\dagger,(n)} = m_{i_2} \ \Big| \ W_k^{\dagger,(n)} = m_{i_1}\right)$$
$$\leq \epsilon + \exp\left(- (1-\epsilon)q_{i_1}u\right)\frac{\nu_{i_1,i_2} + \epsilon}{q_{i_1}} \tag{H.84}$$

- On the other hand, for all $k \geq 0$, all $m_{i_1} \in G^{\text{large}}$, the following claim holds for all $n$ sufficiently large:

$$- \epsilon + \exp\left(- (1+\epsilon)q_{i_1}u\right)\frac{-\epsilon + \sum_{i_2:\, m_{i_2} \notin G} \nu_{i_1,i_2}}{q_{i_1}}$$
$$\leq \mathbb{P}_x\left(\widetilde{S}_{k+1}^{\dagger,(n)} - S_k^{\dagger,(n)} > u, \ \widetilde{W}_{k+1}^{\dagger,(n)} = \dagger \ \Big| \ W_k^{\dagger,(n)} = m_{i_1}\right)$$
$$\leq \mathbb{P}_x\left(\widetilde{S}_{k+1}^{\dagger,(n)} - S_k^{\dagger,(n)} > u - \eta_n^\theta, \ \widetilde{W}_{k+1}^{\dagger,(n)} = \dagger \ \Big| \ W_k^{\dagger,(n)} = m_{i_1}\right)$$
$$\leq \epsilon + \exp\left(- (1-\epsilon)q_{i_1}u\right)\frac{\epsilon + \sum_{i_2:\, m_{i_2} \notin G} \nu_{i_1,i_2}}{q_{i_1}} \tag{H.85}$$

- Here is one implication of eq. (H.81)eq. (H.82). Since $|G| \leq n_{\min}$, we have (when $n$ is sufficiently large)

$$\mathbb{P}_x\left(S_k^{(n)} - \widetilde{S}_k^{(n)} \geq \eta_n^\theta \ \Big| \ \widetilde{W}_k^{(n)} = m_{i_1}\right) < n_{\min} \cdot \epsilon \tag{H.86}$$

for all $k \geq 0$ and $m_{i_1} \in G^{\text{small}}$.

- Note that for any $m_{i_1}, m_{i_2} \in G^{\text{large}}$ and any $k \geq 0$

$$\mathbb{P}_x\left(S_{k+1}^{\dagger,(n)} - S_k^{\dagger,(n)} > u, \ W_{k+1}^{\dagger,(n)} = m_{i_2} \ \Big| \ W_k^{\dagger,(n)} = m_{i_1}\right)$$
$$= \mathbb{1}\{i_2 \neq i_1\}\mathbb{P}_x\left(\widetilde{S}_{k+1}^{\dagger,(n)} - S_k^{\dagger,(n)} > u, \ \widetilde{W}_{k+1}^{\dagger,(n)} = m_{i_2} \ \Big| \ W_k^{\dagger,(n)} = m_{i_1}\right)$$
$$+ \sum_{i_3:\, m_{i_3} \in G^{\text{small}}} \int_{s>0} \mathbb{P}_x\left(S_{k+1}^{\dagger,(n)} - \widetilde{S}_{k+1}^{\dagger,(n)} \geq (u-s) \vee 0, \ \widetilde{W}_{k+1}^{\dagger,(n)} = m_{i_2} \ \Big| \ \widetilde{W}_k^{\dagger,(n)} = m_{i_3}\right)$$
$$\cdot \mathbb{P}_x\left(\widetilde{S}_{k+1}^{\dagger,(n)} - S_k^{\dagger,(n)} = ds, \ \widetilde{W}_{k+1}^{\dagger,(n)} = m_{i_3} \ \Big| \ W_k^{\dagger,(n)} = m_{i_1}\right).$$

Fix some $i_3$ with $m_{i_3} \in G^{\text{small}}$. Due to eq. (H.86)

$$\int_{s \in (0, u-\eta_n^\theta]} \mathbb{P}_x\left(S_{k+1}^{\dagger,(n)} - \widetilde{S}_{k+1}^{\dagger,(n)} \geq u - s, \ \widetilde{W}_{k+1}^{\dagger,(n)} = m_{i_2} \ \Big| \ \widetilde{W}_k^{\dagger,(n)} = m_{i_3}\right)$$
$$\cdot \mathbb{P}_x\left(\widetilde{S}_{k+1}^{\dagger,(n)} - S_k^{\dagger,(n)} = ds, \ \widetilde{W}_{k+1}^{\dagger,(n)} = m_{i_3} \ \Big| \ W_k^{\dagger,(n)} = m_{i_1}\right)$$
$$\leq n_{\min}\epsilon.$$

Meanwhile, by considering the integral on $(u - \eta_n^\theta, \infty)$, we get

$$\int_{s \in (u-\eta_n^\theta, \infty)} \mathbb{P}_x \left( S_{k+1}^{\dagger,(n)} - \widetilde{S}_{k+1}^{\dagger,(n)} \geq (u-s) \vee 0, \ \widetilde{W}_{k+1}^{\dagger,(n)} = m_{i_2} \ \Big| \ \widetilde{W}_k^{\dagger,(n)} = m_{i_3} \right)$$

$$\cdot \mathbb{P}_x \left( \widetilde{S}_{k+1}^{\dagger,(n)} - S_k^{\dagger,(n)} = ds, \ \widetilde{W}_{k+1}^{\dagger,(n)} = m_{i_3} \ \Big| \ W_k^{\dagger,(n)} = m_{i_1} \right)$$

$$\geq \int_{s \in (u,\infty)} \mathbb{P}_x \left( \widetilde{W}_{k+1}^{\dagger,(n)} = m_{i_2} \ \Big| \ \widetilde{W}_k^{\dagger,(n)} = m_{i_3} \right)$$

$$\cdot \mathbb{P}_x \left( \widetilde{S}_{k+1}^{\dagger,(n)} - S_k^{\dagger,(n)} = ds, \ \widetilde{W}_{k+1}^{\dagger,(n)} = m_{i_3} \ \Big| \ W_k^{\dagger,(n)} = m_{i_1} \right)$$

$$\geq (p_{i_3,i_2} - \epsilon) \left( -\epsilon + \exp\left( -(1+\epsilon) q_{i_1} u \right) \frac{\nu_{i_1,i_3} - \epsilon}{q_{i_1}} \right)$$

due to eq. (H.81) and eq. (H.84). As for the upper bound,

$$\int_{s \in (u-\eta_n^\theta, \infty)} \mathbb{P}_x \left( S_{k+1}^{\dagger,(n)} - \widetilde{S}_{k+1}^{\dagger,(n)} \geq (u-s) \vee 0, \ \widetilde{W}_{k+1}^{\dagger,(n)} = m_{i_2} \ \Big| \ \widetilde{W}_k^{\dagger,(n)} = m_{i_3} \right)$$

$$\cdot \mathbb{P}_x \left( \widetilde{S}_{k+1}^{\dagger,(n)} - S_k^{\dagger,(n)} = ds, \ \widetilde{W}_{k+1}^{\dagger,(n)} = m_{i_3} \ \Big| \ W_k^{\dagger,(n)} = m_{i_1} \right)$$

$$\leq (n_{\min}\epsilon + p_{i_3,i_2} + \epsilon) \left( \epsilon + \exp\left( -(1-\epsilon) q_{i_1} u \right) \frac{\nu_{i_1,i_3} + \epsilon}{q_{i_1}} \right)$$

due to eq. (H.81), eq. (H.84) and eq. (H.86).

- Therefore, for any $m_{i_1}, m_{i_2} \in G^{\text{large}}$ and any $k \geq 0$,

$$\mathbb{P}_x \left( S_{k+1}^{\dagger,(n)} - S_k^{\dagger,(n)} > u, \ W_{k+1}^{\dagger,(n)} = m_{i_2} \ \Big| \ W_k^{\dagger,(n)} = m_{i_1} \right)$$

$$\leq g(\epsilon) + \exp\left( -(1-\epsilon) q_{i_1} u \right) \frac{\mathbb{1}\{i_2 \neq i_1\} \nu_{i_1,i_2} + \sum_{i_3: \ m_{i_3} \in G^{\text{small}}} \nu_{i_1,i_3} p_{i_3,i_2}}{q_{i_1}}$$

$$\leq g(\epsilon) + \exp\left( -(1-\epsilon) q_{i_1} u \right) \frac{q_{i_1,i_2}}{q_{i_1}} \tag{H.87}$$

and

$$\mathbb{P}_x \left( S_{k+1}^{\dagger,(n)} - S_k^{\dagger,(n)} > u, \ W_{k+1}^{\dagger,(n)} = m_{i_2} \ \Big| \ W_k^{\dagger,(n)} = m_{i_1} \right)$$

$$\geq -g(\epsilon) + \exp\left( -(1+\epsilon) q_{i_1} u \right) \frac{\mathbb{1}\{i_2 \neq i_1\} \nu_{i_1,i_2} + \sum_{i_3: \ m_{i_3} \in G^{\text{small}}} \nu_{i_1,i_3} p_{i_3,i_2}}{q_{i_1}}$$

$$\geq -g(\epsilon) + \exp\left( -(1+\epsilon) q_{i_1} u \right) \frac{q_{i_1,i_2}}{q_{i_1}} \tag{H.88}$$

where $q^* = \max_i q_i$ and

$$g(\epsilon) \triangleq 2\epsilon + \frac{\epsilon}{q^*} + n_{\min}(1+\epsilon)\epsilon + \epsilon \frac{1+\epsilon}{q^*} n_{\min}$$

$$+ n_{\min}(\epsilon + \frac{\epsilon}{q^*}) + n_{\min}(n_{\min}\epsilon + \epsilon + 1)(1 + \frac{1}{q^*})\epsilon.$$

Note that $\lim_{\epsilon \downarrow 0} g(\epsilon) = 0$.

- On the other hand, for the case where the marker process $\hat{X}^{\dagger,(n)}$ jumps to the cemetery state $\dagger$ from some $m_{i_1} \in G^{\text{large}}$, note that

$$\mathbb{P}_x \left( S_{k+1}^{\dagger,(n)} - S_k^{\dagger,(n)} > u, \ W_{k+1}^{\dagger,(n)} = \dagger \ \Big| \ W_k^{\dagger,(n)} = m_{i_1} \right)$$

$$= \mathbb{P}_x \left( \widetilde{S}_{k+1}^{\dagger,(n)} - S_k^{\dagger,(n)} > u, \ \widetilde{W}_{k+1}^{\dagger,(n)} = \dagger \ \Big| \ W_k^{\dagger,(n)} = m_{i_1} \right)$$

$$+ \sum_{i_3: \ m_{i_3} \in G^{\text{small}}} \int_{s>0} \mathbb{P}_x \left( S_{k+1}^{\dagger,(n)} - \widetilde{S}_{k+1}^{\dagger,(n)} \geq (u-s) \vee 0, \ \widetilde{W}_{k+1}^{\dagger,(n)} = \dagger \ \Big| \ \widetilde{W}_k^{\dagger,(n)} = m_{i_3} \right)$$

$$\cdot \mathbb{P}_x \left( \widetilde{S}_{k+1}^{\dagger,(n)} - S_k^{\dagger,(n)} = ds, \ \widetilde{W}_{k+1}^{\dagger,(n)} = m_{i_3} \ \Big| \ W_k^{\dagger,(n)} = m_{i_1} \right).$$

Arguing similarly as we did above by considering the integral on $[0, u - \eta_n^\theta]$ and $(u - \eta_n^\theta, \infty)$ separately, and using eq. (H.82) and eq. (H.85), we then get (for all $n$ sufficiently large)

$$\mathbb{P}_x\left(S_{k+1}^{\dagger,(n)} - S_k^{\dagger,(n)} > u, \, W_{k+1}^{\dagger,(n)} = \dagger \,\Big|\, W_k^{\dagger,(n)} = m_{i_1}\right)$$

$$\leq g(\epsilon) + \exp\left(-(1-\epsilon)q_{i_1}u\right)\frac{\sum_{i_2:m_{i_2}\notin G}\nu_{i_1,i_2} + \sum_{i_2:m_{i_2}\in G^{\text{small}}}\nu_{i_1,i_2}p_{i_2,\dagger}}{q_{i_1}}$$

$$= g(\epsilon) + \exp\left(-(1-\epsilon)q_{i_1}u\right)\frac{q_{i_1,\dagger}}{q_{i_1}} \tag{H.89}$$

and

$$\mathbb{P}_x\left(S_{k+1}^{\dagger,(n)} - S_k^{\dagger,(n)} > u, \, W_{k+1}^{\dagger,(n)} = \dagger \,\Big|\, W_k^{\dagger,(n)} = m_{i_1}\right)$$

$$\geq -g(\epsilon) + \exp\left(-(1+\epsilon)q_{i_1}u\right)\frac{\sum_{i_2:m_{i_2}\notin G}\nu_{i_1,i_2} + \sum_{i_2:m_{i_2}\in G^{\text{small}}}\nu_{i_1,i_2}p_{i_2,\dagger}}{q_{i_1}}$$

$$= -g(\epsilon) + \exp\left(-(1+\epsilon)q_{i_1}u\right)\frac{q_{i_1,\dagger}}{q_{i_1}} \tag{H.90}$$

For simplicity of presentation, we also let $q_{j,0} = q_{j,\dagger}$ and $p_{j,0} = p_{j,\dagger}$. First of all, remember that we have fixed some series of strictly positive real numbers $(s_k)_{k=0}^K$, some sequence $(w_k)_{k=0}^K \in (\bar{G})^{K+1}$ such that $w_j \neq \dagger$ for any $j < K$, and indices $i_k$ such that $w_k = m_{i_k}$ for each $k$. The definition of the continuous-time Markov chain $Y^\dagger$ implies that

$$\mathbb{P}\left(S_0 < t_0, W_0 = w_0; \, S_k > s_k \text{ and } W_k = w_k \,\forall k \in [K]\right)$$

$$= \mathbb{P}(\pi_G(m_i) = w_0)\prod_{k=1}^K\left(S_k > s_k \text{ and } W_k = m_k \,\Big|\, W_{k-1} = w_{k-1}\right)$$

$$= \left(\mathbb{1}\{m_i \in G^{\text{large}}, \, i_0 = i\} + \mathbb{1}\{m_i \in G^{\text{small}}\}p_{i_0,i_1}\right) \cdot \prod_{k=1}^K \exp(-q_{i_{k-1}}s_k)\frac{q_{i_{k-1},i_k}}{q_{i_{k-1}}}.$$

On the other hand, using eq. (H.87)-eq. (H.90), we know that for all $n$ sufficiently large,

$$\mathbb{P}_x\left(S_0^{(n)} < s_0, W_0^{(n)} = w_0; \, S_k^{(n)} > s_k \text{ and } W_k^{(n)} = w_k \,\forall k \in [K]\right)$$

$$\geq (1-\epsilon)\left(\mathbb{1}\{m_i \in G^{\text{large}}, \, i_0 = i\} + \mathbb{1}\{m_i \in G^{\text{small}}\}(p_{i_0,i_1} - \epsilon)\right)$$

$$\cdot \prod_{k=1}^K\left(-g(\epsilon) + \exp(-(1+\epsilon)q_{i_{k-1}}s_k)\frac{q_{i_{k-1},i_k}}{q_{i_{k-1}}}\right)$$

and

$$\mathbb{P}_x\left(S_0^{(n)} < s_0, W_0^{(n)} = w_0; \, S_k^{(n)} > s_k \text{ and } W_k^{(n)} = m_k \,\forall k \in [K]\right)$$

$$\leq \left(\mathbb{1}\{m_i \in G^{\text{large}}, \, i_0 = i\} + \mathbb{1}\{m_i \in G^{\text{small}}\}(p_{i_0,i_1} + \epsilon)\right)$$

$$\cdot \prod_{k=1}^K\left(g(\epsilon) + \exp(-(1-\epsilon)q_{i_{k-1}}s_k)\frac{q_{i_{k-1},i_k}}{q_{i_{k-1}}}\right).$$

The arbitrariness of $\epsilon > 0$ then allows us to establish eq. (H.79) and conclude the proof. $\qquad\square$

Now we are ready to prove Lemma H.4 and Lemma H.5.

*Proof of Lemma H.4.* From Lemma H.14, one can see the existence of some $(\Delta_n)_{n \geq 1}$ with $\lim_n \Delta_n = 0$ such that eq. (H.60)-eq. (H.63) hold. For simplicity of notations, we let $\hat{X}^{(n)} = \hat{X}^{*,\eta_n,\Delta_n}, X^{(n)} = X^{*,\eta_n}$, and let $\bar{t} = t_{k'}$

Combine eq. (H.60) with Lemma H.13, and we immediately get eq. (H.23). In order to prove eq. (H.24), it suffices to show that for any $\epsilon > 0$,

$$\limsup_n \mathbb{P}_x\Big(X_{t_k}^{(n)} \notin \bigcup_{j:\, m_j \in G^{\text{large}}} B(m_j, \Delta_n)\Big) \leq 4\epsilon \ \forall k \in [k'].$$

Fix $\epsilon > 0$, and observe following bound by decomposing the events

$$\mathbb{P}_x\Big(X_{t_k}^{(n)} \notin \bigcup_{j:\, m_j \in G^{\text{large}}} B(m_j, \Delta_n)\Big)$$

$$\leq \mathbb{P}_x\Big(X_{t_k}^{(n)} \notin \bigcup_{j:\, m_j \in G^{\text{large}}} B(m_j, \Delta_n),\ X_t^{(n)} \in \bigcup_{j:\, m_j \in G} \Omega_j \ \forall t \in [0, \bar{t}]\Big)$$

$$+ \mathbb{P}_x\Big(\exists t \in [0, \bar{t}] \text{ such that } X_t^{(n)} \notin \bigcup_{j:\, m_j \in G} \Omega_j\Big)$$

Therefore, given eq. (H.63), it suffices to prove

$$\limsup_n \mathbb{P}_x\Big(\exists t \in [0, \bar{t}] \text{ such that } X_t^{(n)} \notin \bigcup_{j:\, m_j \in G} \Omega_j\Big) \leq 3\epsilon. \tag{H.91}$$

Let

$$T_0^{(n)} \triangleq \min\{t \geq 0 :\ X_t^{(n)} \in \bigcup_{j: m_j \in G} B(m_j, 2\Delta_n)\}$$

$$I_0^{(n)} = j \iff X_{T_0^{(n)}}^{(n)} \in B(m_j, 2\Delta_n)$$

$$T_k^{(n)} \triangleq \min\{t > T_{k-1}^{(n)} :\ X_t^{(n)} \in \bigcup_{j: m_j \in G,\ j \neq I_{k-1}^{(n)}} B(m_j, 2\Delta_n)\}$$

$$I_k^{(n)} = j \iff X_{T_k^{(n)}}^{(n)} \in B(m_j, 2\Delta_n).$$

Building upon this definition, we define the following stopping times and marks that only records the hitting time to minimizer in *large* attraction fields in $G$ (with convention $\mathbf{k}^{(n),\text{large}}(-1) = -1, T_{-1}^{(n),\text{large}} = 0, T_{-1}^{(n)} = 0$)

$$\mathbf{k}^{(n),\text{large}}(k) \triangleq \min\{l > \mathbf{k}^{(n),\text{large}}(k-1) :\ m_{I_l^{(n)}} \in G^{\text{large}}\}$$

$$T_k^{(n),\text{large}} \triangleq T_{\mathbf{k}^{(n),\text{large}}(k)}^{(n)}, \qquad I_k^{(n),\text{large}} \triangleq I_{\mathbf{k}^{(n),\text{large}}(k)}^{(n)}.$$

Now by defining

$$J^{(n)}(t) \triangleq \#\{k \geq 0 :\ T_k^{(n)} \leq t\},$$

$$J_{\text{large}}^{(n)}(t) \triangleq \#\{k \geq 0 :\ T_k^{(n),\text{large}} \leq t\},$$

$$J^{(n)}(s, t) \triangleq \#\{k \geq 0 :\ T_k^{(n)} \in [s, t]\},$$

we use $J^{(n)}(t)$ to count the numbers of visits to local minima on $G$, and $J_{\text{large}}^{(n)}(t)$ for the number of visits to minimizers in the *large* attraction fields on $G$. $J^{(n)}(s, t)$ counts the indices $k$ such that at $T_k^{(n)}$ a minimizer on $G$ is visited and regarding the hitting time we have $T_k^{(n)} \in [s, t]$.

First of all, the weak convergence result in eq. (H.60) implies the existence of some positive integer $N(\epsilon)$ such that

$$\limsup_n \mathbb{P}_x(J_{\text{large}}^{(n)}(\bar{t}) > N(\epsilon)) < \epsilon.$$

Fix such $N(\epsilon)$. Next, from eq. (H.62), we know the existence of some integer $K(\epsilon)$ such that

$$\limsup_n \sup_{k \geq 0} \mathbb{P}_x\Big(J^{(n)}\big(T_{k-1}^{(n),\text{large}}, T_k^{(n),\text{large}}\big) > K(\epsilon)\Big) \leq \epsilon/N(\epsilon).$$

Fix such $K(\epsilon)$ as well. From the results above, we know that for event

$$A_1(n) \triangleq \{J_{\text{large}}^{(n)}(\bar{t}) \leq N(\epsilon)\} \cap \left\{J^{(n)}\big(T_{k-1}^{(n),\text{large}}, T_k^{(n),\text{large}}\big) \leq K(\epsilon) \; \forall k \in [N(\epsilon)]\right\},$$

we have $\limsup_n \mathbb{P}_x\Big(\big(A_1(n)\big)^c\Big) \leq 2\epsilon$. On the other hand, on event $A_1(n)$, we must have

$$J^{(n)}(\bar{t}) \leq N(\epsilon)K(\epsilon).$$

Meanwhile, it follows immediately from eq. (H.61) that

$$\limsup_n \sup_{k \geq 0} \mathbb{P}\Big(\exists t \in [T_{k-1}^{(n)}, T_k^{(n)}] \text{ such that } X_t^{(n)} \notin \bigcup_{j: \, m_j \in G} \Omega_j\Big) < \frac{\epsilon}{N(\epsilon)K(\epsilon)},$$

hence for event

$$A_2(n) \triangleq \Big\{X_t^{(n)} \in \bigcup_{j: \, m_j \in G} \Omega_j \; \forall t \in [0, T_{N(\epsilon)K(\epsilon)}^{(n)}]\Big\},$$

we must have $\limsup_n \mathbb{P}_x\Big(\big(A_2(n)\big)^c\Big) \leq \epsilon$. To conclude the proof, note that

$$\begin{aligned}
A_1(n) \cap A_2(n) &\subseteq \{J^{(n)}(\bar{t}) \leq N(\epsilon)K(\epsilon)\} \cap \Big\{X_t^{(n)} \in \bigcup_{j: \, m_j \in G} \Omega_j \; \forall t \in [0, T_{N(\epsilon)K(\epsilon)}^{(n)}]\Big\} \\
&= \{T_{N(\epsilon)K(\epsilon)}^{(n)} \geq \bar{t}\} \cap \Big\{X_t^{(n)} \in \bigcup_{j: \, m_j \in G} \Omega_j \; \forall t \in [0, T_{N(\epsilon)K(\epsilon)}^{(n)}]\Big\} \\
&\subseteq \Big\{X_t^{(n)} \in \bigcup_{j: \, m_j \in G} \Omega_j \; \forall t \in [0, \bar{t}]\Big\}
\end{aligned}$$

so we have established eq. (H.91). $\qquad\square$

*Proof of Lemma H.5.* From Lemma H.15, one can see the existence of some $(\Delta_n)_{n \geq 1}$ with $\lim_n \Delta_n = 0$ such that eq. (H.76) and eq. (H.77) hold. For simplicity of notations, we let $\hat{X}^{\dagger,(n)} = \hat{X}^{\dagger,*,\eta_n,\Delta_n}, X^{\dagger,(n)} = X^{\dagger,*,\eta_n}$, and let $\bar{t} = t_{k'}$

Combine eq. (H.76) with Lemma H.13, and we immediately get eq. (H.27). In order to prove eq. (H.28), note that

$$\begin{aligned}
&\Big\{X_{t_k}^{\dagger,(n)} \notin \bigcup_{j: \, m_j \in G^{\text{large}}} B(m_j, \Delta_n) \text{ and } X_{t_k}^{\dagger,(n)} \neq \dagger\Big\} \\
&= \Big\{X_{t_k}^{\dagger,(n)} \notin \bigcup_{j: \, m_j \in G^{\text{large}}} B(m_j, \Delta_n) \text{ and } X_s^{\dagger,(n)} \in \bigcup_{j: m_j \in G} \Omega_j \; \forall s \in [0, t_k]\Big\}
\end{aligned}$$

so the conclusion of the proof follows directly from eq. (H.77). $\qquad\square$

# I  FIRST EXIT TIME OF TRUNCATED HEAVY-TAILED SGD IN $\mathbb{R}^d$

## I.1  MAIN RESULT

The object of interests is the following truncated heavy-tailed SGD iterates

$$X_{k+1}^\eta(x) = X_k^\eta(x) + \varphi_b\big(-\eta\nabla f(X_k^\eta(x)) + \eta Z_{k+1}\big) \; \forall k \geq 0$$

where the initial condition is prescribed by $X_0(x) = x$, $f$ is a real-valued function on $\mathbb{R}^d$, $\eta > 0$ is the learning rate, $(Z_k)_{k \geq 1}$ is the sequence of heavy-tailed noises, the standard gradient clipping operator is $\varphi_b(v) = \min\{1, \frac{b}{\|v\|}\} \cdot v$ with $\|\cdot\|$ being $L_2$ norm. To ease notations, we also use $\mathbb{P}_x$ to denote the conditional law on $\{X_0^\eta = x\}$.

Specifically, we are interested in the first exit time of $X_n^\eta$ from a domain $\mathcal{G}$, i.e. the stopping time

$$\sigma(\eta) = \min\{n \geq 0 : X_n^\eta \notin \mathcal{G}\}.$$

We work with following assumptions.

**Assumption I.1.** *The region $\mathcal{G}$ is connected, bounded and open, and $\boldsymbol{0} \in \mathcal{G}$.*

**Assumption I.2.** *The function $f$ is smooth, i.e. $f \in C^2(\mathbb{R}^d)$.*

**Assumption I.3.** *The boundary set $\partial\mathcal{G}$ is a $(n-1)$-dimensional manifold of class $C^2$ such that the vector field $n(\cdot)$ of the outer normals on $\partial\mathcal{G}$ exists with*

$$\nabla f(v)^T n(v) \geq c_0 \quad \forall v \in \partial\mathcal{G} \tag{I.1}$$

*for some constant $c_0 > 0$.*

**Assumption I.4.** *For $\nabla^2 f(\boldsymbol{0})$, the Hessian of $f(\cdot)$ at point $\boldsymbol{0}$, all the eigenvalues are strictly positive.*

For any $x \in \bar{\mathcal{G}}$, let $\boldsymbol{x}_t(x)$ be the ODE flow with $\boldsymbol{x}_0(x) = x$ solving

$$\dot{\boldsymbol{x}}_t(x) = -\nabla f\big(\boldsymbol{x}_t(x)\big) \quad \forall t \geq 0,$$

**Assumption I.5.** *W.L.O.G., the origin $\boldsymbol{0}$ is an attractor of the domain, i.e. $\nabla f(\boldsymbol{0}) = \boldsymbol{0}$ and $\boldsymbol{0}$ is asymptotically stable in $\mathcal{G}$ in the sense that*

$$\lim_{t\to\infty} \boldsymbol{x}_t(x) = \boldsymbol{0} \quad \forall x \in \bar{\mathcal{G}}.$$

We have the following assumption regarding the heavy-tailed noises $(Z_n)_{n\geq 1}$. For any $x \in \mathbb{R}^d, x \neq 0$, define the following polar transformation

$$\mathbf{T}(x) = (\|x\|, x/\|x\|) \tag{I.2}$$

with $\mathbf{T}_r(x) = \|x\|$, $\mathbf{T}_\theta(x) = x/\|x\|$. Also, let $\mathbb{O} = \{\boldsymbol{0}\}$

**Assumption I.6.** *$\mathbb{E}Z_1 = \boldsymbol{0}$. Besides, there exist a positive integer $m \geq 1$, a sequence $1 < \alpha_1 < \alpha_2 < \cdots < \alpha_m < \infty$, a sequence of slowly-varying functions $(l_1, \cdots, l_m)$, a sequence of probability measures $(S_1, \cdots, S_m)$ on the unit sphere $\mathbb{S}^{d-1}$ with support $F_j \triangleq supp(S_j)$ as closed sets on $\mathbb{S}^{d-1}$ such that*

- *$F_i \cap F_j = \emptyset$ for any $i \neq j$;*

- *For any $j \in [m]$, define the cone $E_j \triangleq \boldsymbol{T}_\theta^{-1}(F_j) \cup \mathbb{O}$ in $\mathbb{R}^d$ and measure $\mathbb{P}^{(j)}(\cdot) = \mathbb{P}(Z_1 \in \cdot \cap E_j)$, we have*

$$t^{\alpha_j} \cdot l_j(t) \cdot \mathbb{P}^{(j)} \circ \boldsymbol{T}^{-1}\big(t \cdot dr \times d\theta\big) \to \nu_{\alpha_j}(dr) \times S_j(d\theta) \quad as\ t \to \infty \tag{I.3}$$

  *in the sense of $\mathbb{M}(E_j \backslash \mathbb{O})$. Here $\nu_\alpha$ is a Borel measure defined on $(0, \infty)$ satisfying $\nu_\alpha[t, \infty) = t^{-\alpha} \ \forall t > 0$;*

- *For measure $\mathbb{P}^{(0)}(\cdot) \triangleq \mathbb{P}\big(Z_1 \in \cdot \backslash(\cup_{j=1}^m E_j)\big)$ and any $\alpha > 0$,*

$$t^\alpha \cdot \mathbb{P}^{(0)} \circ \boldsymbol{T}^{-1}\big(t \cdot dr \times d\theta\big) \to 0 \quad as\ t \to \infty \tag{I.4}$$

  *in the sense of $\mathbb{M}(\mathbb{R}^d \backslash \mathbb{O})$;*

- *For any $j \in [m]$, the measure $S_j$ is absolutely continuous w.r.t. spherical measure on $\mathbb{S}^{d-1}$.*

A function $l : \mathbb{R}_+ \mapsto \mathbb{R}_+$ is slowly varying (at $\infty$) if $\lim_{t\to\infty} l(tx)/l(x) = 1$ holds for any $x > 0$. For details on $\mathbb{M}-$convergence and regular variation in general metric spaces, see Lindskog et al. (2014). Here we state one implication of the assumption. For any $j = 0, 1, \cdots, m$, let

$$H_j(x) \triangleq \mathbb{P}_j\Big(\{\boldsymbol{y} \in \mathbb{R}^d : \|\boldsymbol{y}\| \geq x\}\Big). \tag{I.5}$$

The assumption above immediately implies that, for any $j = 1, \cdots, m$, $H_j(\cdot)$ is regularly varying (at $\infty$) with index $-\alpha_j$ (denoted as $H_j \in RV_{-\alpha_j}$), namely $\lim_{t\to\infty} H_j(tx)/H_j(t) = x^{-\alpha_j} \ \forall x > 0$. Meanwhile, for any $\alpha > 0$, $H_0(x) = o(1/x^\alpha)$ as $x \to \infty$.

The asymptotically behavior of the first exit time hinges on the following geometric characterization of the domain $\mathcal{G}$. For any integer $k \geq 1$, a sequence of strictly positive real numbers $(t_2, \cdots, t_k)$ with $t_1 = 0$, a sequence of non-zero vectors $(w_1, \cdots, w_k)$ and some $\eta > 0$, let

$\boldsymbol{t}^{(k)} = (t_1, \cdots, t_k), \boldsymbol{w}^{(k)} = (w_1, \cdots, w_k)$, and define the ODE path with $k$ jumps (clipped at size $b$) by $\boldsymbol{w}^{(k)} = (w_1, \cdots, w_k)$, we define perturbed ODE path $\widetilde{\boldsymbol{x}}^\eta$ as

$$\widetilde{\boldsymbol{x}}^\eta(0; \boldsymbol{t}^{(k)}, \boldsymbol{w}^{(k)}) = \varphi_b(\eta w_1);$$

$$\frac{d\widetilde{\boldsymbol{x}}^\eta(t, x; \boldsymbol{t}^{(k)}, \boldsymbol{w}^{(k)})}{dt} = -\eta \nabla f\big(\widetilde{\boldsymbol{x}}^\eta(t, x; \boldsymbol{t}^{(k)}, \boldsymbol{w}^{(k)})\big) \quad \forall t \notin \{t_1, t_1 + t_2, \cdots, \sum_{j=1}^k t_j\}$$

$$\widetilde{\boldsymbol{x}}^\eta(t, x; \boldsymbol{t}^{(k)}, \boldsymbol{w}^{(k)}) = \widetilde{\boldsymbol{x}}^\eta(t-, x; \boldsymbol{t}^{(k)}, \boldsymbol{w}^{(k)}) + \varphi_b(\eta w_j) \quad \text{if } t = \sum_{i=1}^j t_i \text{ for some } j$$

with the convention that $g(t-) \triangleq \lim_{s \uparrow t} g(s)$ for any function $g$. Also, when $\eta = 1$ we simply write $\widetilde{\boldsymbol{x}}$. Now we can assign a *cost* to each jump $w_j$ based on the direction using the following function:

$$J(w) = \begin{cases} \alpha_j - 1 & \text{if } w \neq 0, w \in E_j, \\ \infty & \text{otherwise.} \end{cases} \tag{I.6}$$

Given any set of perturbations described by $\boldsymbol{t}^{(k)} \in \{0\} \times \mathbb{R}_+^{k-1}, \boldsymbol{w}^{(k)} \in (\mathbb{R}^d \backslash \mathbb{O})^k$, we identify the *destination* of the flow as

$$h(k; \boldsymbol{t}^{(k)}, \boldsymbol{w}^{(k)}) = \widetilde{\boldsymbol{x}}(\sum_{j=1}^k t_k; \boldsymbol{t}^{(k)}, \boldsymbol{w}^{(k)}).$$

Besides, define function $\mathcal{I}(\boldsymbol{w}^{(k)}) = (i_1, \cdots, i_m)$ if $\#\{i \in [k] : w_i \in E_j\} = i_j$ for all $j \in [m]$. This allows us to define the following configuration sets

$$\mathcal{A}(i_1, \cdots, i_m) \triangleq \big\{\boldsymbol{w}^{(k)} \in (\mathbb{R}^d \backslash \mathbb{O})^k : \mathcal{I}(\boldsymbol{w}^{(k)}) = (i_1, \cdots, i_m), \ k = \sum_{j=1}^m i_j\big\} \tag{I.7}$$

for any $(i_1, \cdots, i_m) \in \mathbb{N}^m$, i.e., some set of jumps $\boldsymbol{w}^{(k)}$ is said to have configuration $(i_1, \cdots, i_m)$ or belong to the configuration set $\mathcal{A}(i_1, \cdots, i_m)$ if the number of jumps in cone $E_j$ is equal to $i_j$. We can also define the cost for each configuration as

$$\mathcal{J}(i_1, \cdots, i_m) = \sum_{j=1}^m (\alpha_j - 1) i_j. \tag{I.8}$$

Now we can characterize the minimum *cost* to exit $\mathcal{G}$:

$$J_{\mathcal{G}} \triangleq \min\{\sum_{j=1}^k J(w_j) : \exists k \in \mathbb{N}, \boldsymbol{t}^{(k)} \in \{0\} \times \mathbb{R}_+^{k-1}, \boldsymbol{w}^{(k)} \in (\mathbb{R}^d \backslash \mathbb{O})^k$$

$$\text{s.t. } \widetilde{\boldsymbol{x}}(\sum_{j=1}^k t_k, \boldsymbol{0}; \boldsymbol{t}^{(k)}, \boldsymbol{w}^{(k)}) \notin \mathcal{G}\}. \tag{I.9}$$

From the boundedness of $\mathcal{G}$, one can see that $0 < J_{\mathcal{G}} < \infty$ regardless of the actual value of the clipping threshold $b > 0$. We need the following technical assumption regarding the configurations of jumps that can trigger the exit with the minimum cost.

**Assumption I.7.** *There exists only one array $(i_1, \cdots, i_m) \in \mathbb{N}^m$ such that $\sum_{j=1}^m i_j(\alpha_j - 1) = J_{\mathcal{G}}$ and (for $k = \sum_{j=1}^m i_m$)*

$$\exists \boldsymbol{t}^{(k)} \in \{0\} \times \mathbb{R}_+^{k-1}, \boldsymbol{w}^{(k)} \in \mathcal{A}(i_1, \cdots, i_m) \text{ s.t. } \widetilde{\boldsymbol{x}}(\sum_{j=1}^k t_k, \boldsymbol{0}; \boldsymbol{t}^{(k)}, \boldsymbol{w}^{(k)}) \notin \mathcal{G}.$$

We use $\boldsymbol{i}^* = (i_1^*, \cdots, i_m^*)$ to denote the unique configuration in Assumption I.7 and $k^* = \sum_{j=1}^m i_j^*$. The implication is that, for a set of jumps $\boldsymbol{t}^{(k)}, \boldsymbol{w}^{(k)}$ with *any other configuration*, one of the following must happen: (i) this set of jumps has a cost strictly higher than $J_G$; (ii) this set of jumps cannot send the ODE flow out of $\mathcal{G}$.

Meanwhile, we introduce the following concept as the *coverage* of a certain configuration set.

$$\mathcal{G}(i_1, \cdots, i_m) \triangleq \big\{ \widetilde{\boldsymbol{x}}(s, \boldsymbol{0}; \boldsymbol{t}^{(k)}, \boldsymbol{w}^{(k)}) :$$

$$k = \sum_{j=1}^{m} i_j, \ \boldsymbol{t}^{(k)} \in \{0\} \times \mathbb{R}_+^{k-1}, \ \boldsymbol{w}^{(k)} \in \mathcal{A}(i_1, \cdots, i_m), \ s \in [0, \sum_{j=1}^{k} t_j] \big\}. \qquad (\text{I.10})$$

It is easy to see that, for any configuration $(i_1, \cdots, i_m)$ with cost $\mathcal{J}(i_1, \cdots, i_m) \leq J_{\mathcal{G}}$, the coverage $\mathcal{G}(i_1, \cdots, i_m)$ is a closed set, and the following technical assumption holds for (Lebesgue) almost every $b > 0$. Here the distance $\boldsymbol{d}(A, B) = \inf_{x \in A, y \in B} \|x - y\|$ for any $A, B \subseteq \mathbb{R}^d$, and $A^\circ$ is the interior of the set $A$.

**Assumption I.8.** *For any $(i_1, \cdots, i_m) \in \mathbb{N}^m$, one of the following must occur:*

- $\boldsymbol{d}\big(\mathcal{G}(i_1, \cdots, i_m), \mathcal{G}^c\big) > 0$;

- $\big(\mathcal{G}(i_1, \cdots, i_m) \cap \mathcal{G}^c\big)^\circ \neq \emptyset$

Note that with $\boldsymbol{i}^*$ and $k^*$ defined above, we can define the following mapping $h$ from $\boldsymbol{r} = (r_1, \cdots, r_{k^*}), \boldsymbol{\theta} = (\theta_1, \cdots, \theta_{k^*}) \in (\mathbb{S}^{d-1})^{k^*}, \boldsymbol{t} = (t_1, \cdots, t_{k^*}) \in \{0\} \times \mathbb{R}_+^{k^*}$ such that

$$h(\boldsymbol{r}, \boldsymbol{\theta}, \boldsymbol{t}) = \widetilde{\boldsymbol{x}}(\sum_{i=1}^{k^*} t_i, \boldsymbol{0}; \ \boldsymbol{t}, \boldsymbol{w})$$

where $\boldsymbol{w} = (w_i)_{i=1}^{k^*}$ with $w_i = r_i \theta_i$. Also, we introduce the concept of *type* for configuration $\boldsymbol{i}^*$. Specifically, define

$$\boldsymbol{j}(i_1, \cdots, i_m) \triangleq \big\{ (j_1, \cdots, j_{k^*}) \in \{0, 1, 2, \cdots, m\}^{k^*} : \#\{n : j_n = k\} = i_k \ \forall k \in [m] \big\}$$

and for any $\boldsymbol{j} \in \boldsymbol{j}(\boldsymbol{i}^*)$, we say that $\boldsymbol{w} = (w_1, \cdots, w_{k^*}) \in \mathcal{A}(\boldsymbol{i}^*)$ has type $\boldsymbol{j}$ if

$$w_i \in E_{j_i} \ \forall i \in [k^*].$$

In other words, based on the direction of each jump in $\boldsymbol{w} \in \mathcal{A}(\boldsymbol{i}^*)$ we group them into different types. Note that $|\boldsymbol{j}(\boldsymbol{i}^*)| < \infty$. Now for any type $\boldsymbol{j} \in \boldsymbol{j}(\boldsymbol{i}^*)$, define a (Borel) measure on $\mathbb{R}_+^{k^*} \times (\mathbb{S}^{d-1})^{k^*} \times \mathbb{R}_+^{k^*-1}$ as

$$\mu_{\boldsymbol{j}} \triangleq (\prod_{i=1}^{k^*} \nu_{\alpha_{j_i}}) \times (\prod_{i=1}^{k^*} S_{j_i}) \times \boldsymbol{m}_{\text{Leb}}^{k^*-1}$$

where, for any $\alpha > 0$, the measure $\nu_\alpha$ is the Borel measure on $(0, \infty)$ with $\nu_\alpha(x, \infty) = 1/x^{1+\alpha}$. Lastly, define measure $\mu$ as $\mu = \sum_{\boldsymbol{j} \in \boldsymbol{j}(\boldsymbol{i}^*)} \mu_{\boldsymbol{j}}$. As will be established in Lemma K.2, the following technical assumption is also a very moderate one since it holds for (Lebesgue) almost every $b > 0$ under the current setting.

**Assumption I.9.** *The set $h^{-1}\big(\partial\mathcal{G}\big)$ has zero mass under the measure $\mu$.*

Having specified the problem setting, we are now ready to present Theorem I.1, the main result of this section. The implication of the theorem is clear: under proper scaling, the first exit time $\sigma(\eta)$ converges in distribution to an Exponential random variable. Moreover, the scaling $\lambda(\eta)$ is roughly of order $\eta^{1+J_{\mathcal{G}}}$, implying that the first exit time $\sigma(\eta)$ is roughly of order $1/\eta^{1+\mathcal{G}}$ as the learning rate $\eta$ approaches 0.

**Theorem I.1.** *Let Assumptions I.1-I.9 hold. There exists a function $\lambda(\eta)$ that is regularly varying (as $\eta \downarrow 0$) with index $(1 + J_{\mathcal{G}})$ such that, for any $x \in \mathcal{G}$ and any $t > 0$,*

$$\lim_{\eta \downarrow 0} \mathbb{P}_x(\sigma(\eta)\lambda(\eta) > t) = \exp(-qt)$$

*where the constant $q = \mu\big(h^{-1}(\mathcal{G}^c)\big)$.*

The proof is provided in Section J. As a concluding remark, we stress that the order of the first exit time is dictated by $J_{\mathcal{G}}$, the minimum cost for exit we introduced above. Bearing obvious similarity to

the first exit time analysis of SDE driven by heavy-tailed Lévy processes in Imkeller et al. (2010a), our result can be viewed as a natural generalization when gradient clipping is applied and heavy-tailed noises may not always align with a finite number of lines. In Imkeller et al. (2010a), the unclipped setting implies that the escape from the domain $\mathcal{G}$ can always be achieved with one big jump. However, a big perturbation (in noise) along different directions may correspond to different heavy-tailed indices, i.e., induce different costs given our definition of cost function $J(w)$ in eq. (I.6). In Theorem 1 of Imkeller et al. (2010a), we see that the order of the first exit time is determined by $\alpha_1$, the smallest heavy-tailed index. The underlying reason is that the exit is almost always trigger by a single big jump with the smallest cost. Similarly, in our setting where multiple jumps are required for escape due to the clipping mechanism, we see that the order of the first exit time is not dictated by the number of jumps or the accumulated distances of the jumps, but the smallest possible accumulated *cost* defined as the summation of $J(w_i)$ where $w_i$'s are the jumps the lead to escape from $\mathcal{G}$. As detailed in the proof, this is because such jumps with the smallest costs $J_{\mathcal{G}}$ dictates the most likely way for exiting $\mathcal{G}$.

## I.2 A Special Case: Uniform Heavy Tail Index Along All Directions

As stated above, Theorem I.1 deals with general case where the noise distribution is allowed to have different heavy-tailed indices along different directions (see Assumption I.6). As a special case, it is worth noticing that if a single heavy-tailed index $\alpha$ can be used to describe the tail behavior of noises along any direction (this can be easily guaranteed if the heavy-tailed noise is manually injected into a light-tailed setting), then the minimum cost $J_{\mathcal{G}}$ will be equal to $l^*(\alpha - 1)$ where $l^*$ is the minimum number of jumps required for escape. The readers can see that this is a natural extension of our $\mathbb{R}^1$ results, implying that the same strong preference for "wide" minima still hold in $\mathbb{R}^d$ under truncated heavy-tailed SGD. To be specific, we work with the following assumption about the noise distribution.

**Assumption I.10.** $\mathbb{E}Z_1 = \mathbf{0}$. *Besides, there exists some $\alpha > 1$, a slowly-varying functions $l$, a probability measures $S$ on the unit sphere $\mathbb{S}^{d-1}$ with support $supp(S_j) = \mathbb{S}^{d-1}$ such that*

- *For the measure $\nu(\cdot) = \mathbb{P}(Z_1 \in \cdot \cap E_j)$, we have*

$$t^\alpha \cdot l(t) \cdot \nu \circ \boldsymbol{T}^{-1}(t \cdot dr \times d\theta) \to \nu_\alpha(dr) \times S(d\theta) \quad as \ t \to \infty \tag{I.11}$$

  *in the sense of $\mathbb{M}(\mathbb{R}^d \setminus \mathbb{O})$. Here $\nu_\alpha$ is a Borel measure defined on $(0, \infty)$ satisfying $\nu_\alpha[t, \infty) = t^{-\alpha} \ \forall t > 0$;*

- *The measure $S$ is absolutely continuous w.r.t. spherical measure on $\mathbb{S}^{d-1}$.*

Compared to Assumption I.6, one can easily see that Assumption I.10 is a stronger version with the specific proviso that a single index $\alpha$ can describe the tail of the noise distribution along any direction in $\mathbb{R}^d$. The first exit time results now admit a simplified form. In particular, the cost of jumps will degenerate to the *count* of jumps in the sense that $\sum_{i=1}^k J(w_i) = k(\alpha - 1)$ for any $k \geq 1$ and any $(w_1, \cdots, w_k) \in (\mathbb{R}^d \setminus \mathbb{O})^k$. Moreover, we now have $J_{\mathcal{G}} = l_{\mathcal{G}}^* \cdot (\alpha - 1)$ where

$$l_{\mathcal{G}}^* \triangleq \min\{k \in \mathbb{N} : \exists \boldsymbol{t}^{(k)} \in \{0\} \times \mathbb{R}_+^{k-1}, \boldsymbol{w}^{(k)} \in (\mathbb{R}^d \setminus \mathbb{O})^k \ \text{s.t.} \ \widetilde{\boldsymbol{x}}(\sum_{j=1}^k t_k, \mathbf{0}; \boldsymbol{t}^{(k)}, \boldsymbol{w}^{(k)}) \notin \mathcal{G}\}$$

and the measure $\mu$ can now be expressed as $\mu = (\prod_{i=1}^{l_{\mathcal{G}}^*} \nu_\alpha) \times (\prod_{i=1}^{l_{\mathcal{G}}^*} S) \times \boldsymbol{m}_{\text{Leb}}^{l_{\mathcal{G}}^* - 1}$. Therefore, the following theorem is merely a restatement of Theorem I.1 in this simplified setting. Still, we present the result to highlight the role of the minimum jump number $J_{\mathcal{G}}$ in the first exit time.

**Theorem I.2.** *Let Assumptions I.1-I.5, I.10 and I.7-I.9 hold. There exists a function $\lambda(\eta)$ that is regularly varying (as $\eta \downarrow 0$) with index $1 + l_{\mathcal{G}}^*(\alpha - 1)$ such that, for any $x \in \mathcal{G}$ and any $t > 0$,*

$$\lim_{\eta \downarrow 0} \mathbb{P}_x(\sigma(\eta)\lambda(\eta) > t) = \exp(-qt)$$

*where the constant $q = \mu(h^{-1}(\mathcal{G}^c))$.*

It is clear that first exit time since the first exit time is roughly of order $1/\eta^{1+(\alpha-1)l_{\mathcal{G}}^*}$. Therefore, even in the general $\mathbb{R}^d$ case, the quantity $l_{\mathcal{G}}^*$, i.e. the minimum count of jumps to escape from a domain $\mathcal{G}$,

induces a hierarchy of first exit time as the first exit time from the domain with largest $l^*_{\mathcal{G}}$ (requiring most number of jumps to escape) will dominate the first exit time from other regions.

Lastly, if we work under the standard setting such as the ones in Zhou et al. (2020) where local strong convexity of $f$ in the domain $\mathcal{G}$ is assumed, the results can be further simplified and the minimum jump number $l^*_{\mathcal{G}}$ will be directly tied to the *width* of the each domain. Specifically, the quantity $r_{\mathcal{G}} \triangleq \sup_{y \in \mathcal{G}} \|y\|$ can be interpreted the effective width or radius of $\mathcal{G}$, and we work with the following assumption.

**Assumption I.11.** *The function $f$ is strongly convex on the closed ball $\{y \in \mathbb{R}^d : \|y\| \leq r_{\mathcal{G}}\}$.*

Two consequences follow immediately from this assumption. First of all, there exists some constant $c > 0$ such that

$$\|\boldsymbol{x}_t(x)\| \leq \|x\| \, e^{-ct} \ \forall x \in \{y : \ \|y\| < r_{\mathcal{G}}\}.$$

As a result, we must have $l^*_{\mathcal{G}} \geq r_{\mathcal{G}}/b$. Next, as long as $r_{\mathcal{G}}/b$ is not an integer (which holds for Lebesgue almost every $b > 0$), for $k = \lceil r_{\mathcal{G}}/b \rceil$ we can find $t_2 > 0, \cdots, t_k > 0$ and $w_1 \neq \boldsymbol{0}, \cdots, w_k \neq \boldsymbol{0}$ where $\widetilde{\boldsymbol{x}}\big(s, \boldsymbol{0}; (0, t_2, \cdots, t_k), (w_1, \cdots, w_k)\big) \notin \mathcal{G}$ for some $s > 0$. In fact, it is worth noticing that Assumption I.8 now degenerates to the condition that $r_{\mathcal{G}}/b$ is not an integer, and we now know that for any such $b > 0$, we have $l^*_{\mathcal{G}} \leq \lceil r_{\mathcal{G}}/b \rceil$. In summary, we have established the following result indicating that the strong preference for wider minima under truncated heavy-tailed SGD still persists in $\mathbb{R}^d$ given proper convexity assumption on $f$.

**Theorem I.3.** *Let Assumptions I.1-I.5, I.10-I.11 and I.7-I.9 hold. There exists a function $\lambda(\eta)$ that is regularly varying (as $\eta \downarrow 0$) with index $1 + l^*(\alpha - 1)$ such that, for any $x \in \mathcal{G}$ and any $t > 0$,*

$$\lim_{\eta \downarrow 0} \mathbb{P}_x(\sigma(\eta)\lambda(\eta) > t) = \exp(-qt)$$

*where the constant $q = \mu\big(h^{-1}(\mathcal{G}^c)\big)$ and $l^* = \lceil r_{\mathcal{G}}/b \rceil$.*

## I.3 RELAXING THE TECHNICAL ASSUMPTIONS

In order to achieve the tightest characterization of the first exit time, some assumptions introduced above are slightly stronger than the ones in Imkeller et al. (2010a). For instance, in Assumption I.3 we require $\partial \mathcal{G}$ to be of class $C^2$ while the assumption (A3) in Imkeller et al. (2010a) only requires it to be a $C^1$ manifold. Besides, Assumption I.6 requires that $S_j$, the limiting distribution of the directions of each heavy-tailed component in noises, to be absolutely continuous w.r.t. the spherical measure. As will be stressed in Section K.1, these conditions are only imposed to prove Lemma K.2, thus ensuring that Assumption I.9 holds for almost every $b > 0$. Briefly speaking, all the efforts to guarantee Assumption I.9 allows us to conclude that the law of scaled first exit time $\lambda(\eta)\sigma(\eta)$ converges exactly to that of $Exp(q)$ with $q = \mu\big(h^{-1}(\mathcal{G}^c)\big)$.

Fortunately, the discussion below will show that, even when we relax all the extra technical assumptions above (hence removing Assumption I.9), the order of the first exit time is still dictated by the minimum cost for escape $J_{\mathcal{G}}$, and a similar result about the first exit time can be obtained where the distribution of the scaled first exit time $\lambda(\eta)\sigma(\eta)$ will be asymptotically bounded by two Exponential RVs.

Specifically, we reiterate that the first half of Assumption I.3, i.e. $\partial \mathcal{G}$ is a differential manifold, will only be applied to prove Lemma K.2. The second half of the proof, namely the lower bound in eq. (I.1), only serves the ensure that for any $x \in \mathcal{G}$, we have $\boldsymbol{x}_t(x) \in \mathcal{G} \ \forall t \geq 0$ so that the gradient flow starting in $\mathcal{G}$ will never leave this domain. Now let us focus on the open sets $\mathcal{G}^\epsilon$ and $\mathcal{G}_\epsilon$ for some small $\epsilon > 0$ where

$$\mathcal{G}^\epsilon \triangleq \{y \in \mathbb{R}^d : \ \boldsymbol{d}(y, \mathcal{G}) < \epsilon\}, \quad \mathcal{G}_\epsilon \triangleq \{y \in \mathcal{G} : \ \boldsymbol{d}(y, \mathcal{G}^c) > \epsilon\}.$$

Note that $\cap_\epsilon \mathcal{G}^\epsilon = \overline{\mathcal{G}}$, $\cup_\epsilon \mathcal{G}_\epsilon = \mathcal{G}$ and for any positive reals $\epsilon_1, \epsilon_2$ such that $\epsilon_1 \neq \epsilon_2$ and the two reals are small enough for $G_{\epsilon_1}$ and $G_{\epsilon_2}$ to be non-empty,

$$\partial \mathcal{G}^{\epsilon_1} \cap \partial \mathcal{G}^{\epsilon_2} = \emptyset, \quad \partial \mathcal{G}_{\epsilon_1} \cap \partial \mathcal{G}_{\epsilon_2} = \emptyset.$$

Given eq. (I.1), as well as the smoothness of the vector field $\nabla f$ (see Assumption I.2) and the fact that $\mathcal{G}$ is connected, bounded and open and contains a unique attractor $\boldsymbol{0}$ (see Assumption I.1 and I.5),

it is easy to see that for all $\epsilon > 0$ sufficiently small, we will have $\boldsymbol{x}_t(x) \in \mathcal{G}^\epsilon \ \forall t \geq 0$ for any $x \in \mathcal{G}^\epsilon$ and $\boldsymbol{x}_t(x) \in \mathcal{G}_\epsilon \ \forall t \geq 0$ for any $x \in \mathcal{G}_\epsilon$.

Meanwhile, in our proof below we will establish eq. (J.17) (which, again, does not require the $C^2$ class assumption on $\partial \mathcal{G}$ or the absolute continuity assumption on the measures $S_j$). Note that eq. (J.17) implies the existence of $\Delta > 0$ such that $\mu\big(h^{-1}(\mathcal{G}_\Delta)\big) < \infty$. Therefore, for all but only countably many $\epsilon \in (0, \Delta)$, we have $\mu\big(h^{-1}(\partial \mathcal{G}^\epsilon)\big) = \mu\big(h^{-1}(\partial \mathcal{G}_\epsilon)\big) = 0$. In other words, we can find a sequence of $\epsilon_n$ with $\lim_n \epsilon_n = 0$ such that

$$\mu\big(h^{-1}(\partial \mathcal{G}^{\epsilon_n})\big) = \mu\big(h^{-1}(\partial \mathcal{G}_{\epsilon_n})\big) = 0 \ \ \forall n \geq 1$$

. Also, for a fixed $b > 0$ that ensures Assumption I.7 and I.8, one can see that similar conditions will also hold for sets $\mathcal{G}^\epsilon$ and $\mathbb{G}_\epsilon$ as long as $\epsilon > 0$ is sufficiently small. In summary, if we consider first exit times

$$\sigma^n(\eta) \triangleq \min\{k \geq 0 : \ X_k^\eta \notin \mathcal{G}^{\epsilon_n}\}, \quad \sigma_n(\eta) \triangleq \min\{k \geq 0 : \ X_k^\eta \notin \mathcal{G}_{\epsilon_n}\}$$

with the obvious bounds that $\sigma_n(\eta) \leq \sigma(\eta) \leq \sigma^n(\eta)$ for all $n \geq 1$, then it suffices to apply Theorem I.1 directly onto $\sigma_n(\eta), \sigma^n(\eta)$. Note that we did not attempt to establish that $\partial \mathcal{G}^\epsilon$ or $\partial \mathcal{G}_\epsilon$ are differential manifolds. However, we reiterate that this is not needed since the $C^2$ manifold condition in Assumption I.3 only serves to ensure that Assumption I.9 holds for almost surely every $b > 0$, which we sidestep by picking a proper sequence $\epsilon_n$ to explicitly satisfy the condition. More formally speaking, consider the following relaxted assumptions.

**Assumption I.12.** *The boundary set $\partial \mathcal{G}$ is a $(n-1)$-dimensional manifold of class $C^1$ such that the vector field $n(\cdot)$ of the outer normals on $\partial \mathcal{G}$ exists with*

$$\nabla f(v)^T n(v) \geq c_0 \ \ \forall v \in \partial \mathcal{G}$$

*for some constant $c_0 > 0$.*

**Assumption I.13.** $\mathbb{E} Z_1 = \boldsymbol{0}$. *Besides, there exist a positive integer $m \geq 1$, a sequence $0 < \alpha_1 < \alpha_2 < \cdots < \alpha_m < \infty$, a sequence of slowly-varying functions $(l_1, \cdots, l_m)$, a sequence of probability measures $(S_1, \cdots, S_m)$ on the unit sphere $\mathbb{S}^{d-1}$ with support $F_j \triangleq supp(S_j)$ as closed sets on $\mathbb{S}^{d-1}$ such that*

- *$F_i \cap F_j = \emptyset$ for any $i \neq j$;*

- *For any $j \in [m]$, define the cone $E_j \triangleq \boldsymbol{T}_\theta^{-1}(F_j) \cup \mathbb{O}$ in $\mathbb{R}^d$ and measure $\mathbb{P}^{(j)}(\cdot) = \mathbb{P}(Z_1 \in \cdot \cap E_j)$, we have*

  $$t^{\alpha_j} \cdot l_j(t) \cdot \mathbb{P}^{(j)} \circ \boldsymbol{T}^{-1}\big(t \cdot dr \times d\theta\big) \to \nu_{\alpha_j}(dr) \times S_j(d\theta) \ \ as \ t \to \infty$$

  *in the sense of $\mathbb{M}(E_j \backslash \mathbb{O})$. Here $\nu_\alpha$ is a Borel measure defined on $(0, \infty)$ satisfying $\nu_\alpha[t, \infty) = t^{-\alpha} \ \forall t > 0$;*

- *For measure $\mathbb{P}^{(0)}(\cdot) \triangleq \mathbb{P}\big(Z_1 \in \cdot \backslash (\cup_{j=1}^m E_j)\big)$ and any $\alpha > 0$,*

  $$t^\alpha \cdot \mathbb{P}^{(0)} \circ \boldsymbol{T}^{-1}\big(t \cdot dr \times d\theta\big) \to 0 \ \ as \ t \to \infty$$

  *in the sense of $\mathbb{M}(\mathbb{R}^d \backslash \mathbb{O})$.*

We say Assumptions I.1-I.2, I.12, I.4-I.5, I.13, I.7-I.8 are the set of *relaxed assumptions*. The discussion above implies that the following result is an immediate consequence from Theorem I.1.

**Theorem I.4.** *Let the relaxed assumptions hold. There exists a function $\lambda(\eta)$ that is regularly varying (as $\eta \downarrow 0$) with index $(1 + J_\mathcal{G})$ such that, for any $x \in \mathcal{G}$ and any $t > 0$,*

$$\exp(-qt) \leq \liminf_{\eta \downarrow 0} \mathbb{P}_x(\sigma(\eta)\lambda(\eta) > t) \leq \limsup_{\eta \downarrow 0} \mathbb{P}_x(\sigma(\eta)\lambda(\eta) > t) \leq \exp(-q^\uparrow t)$$

*with $q = \mu\big(h^{-1}(\mathcal{G}^c)\big), q^\uparrow = \mu\Big(h^{-1}\big((\overline{\mathcal{G}})^c\big)\Big)$.*

## J  PROOF OF THEOREM I.1

This section is devoted to establishing Theorem I.1. For clarity of the exposition, we break down the proof into several major steps, each contained in a subsection below.

## J.1 Picking constants $\bar{t}, \bar{\epsilon}, \bar{\delta}$

The goal of this subsection is to fix several important constants that characterize the typical sizes and inter-arrival times of large perturbations in SGD trajectory that can cause the escape from domain $\mathcal{G}$. We will abuse the notations slightly when referencing certain constants, and quantities such as $c_0, c_1, c_2$ may not be equal to the ones used in assumptions above.

First, from Assumption I.6 and the definition of $J_{\mathcal{G}}$, we can define

$$l^* = \lceil J_{\mathcal{G}}/(\alpha_1 - 1) \rceil + 1 \tag{J.1}$$

and we must have $\infty > l^* > k^*$. Meanwhile, from Assumption I.8, we can find $\bar{\epsilon} > 0$ such that

$$\boldsymbol{d}\big(\mathcal{G}(i_1, \cdots, i_m), \mathcal{G}^c\big) > 100 l^* \bar{\epsilon} \tag{J.2}$$

for any $(i_1, \ldots, i_m) \in \mathbb{N}^m$ with $\sum_{j=1}^m i_j(\alpha_j - 1) < J_{\mathcal{G}}$, as well as

$$\exists x \in \mathcal{G}(\boldsymbol{i}^*) \text{ such that } \boldsymbol{d}(x, \mathcal{G}) > 100 l^* \bar{\epsilon}.$$

Furthermore, Assumption I.5 implies that $\nabla f(v) \neq \boldsymbol{0}$ for any $v \in \overline{\mathcal{G}} \backslash \{\boldsymbol{0}\}$. From Assumptions I.3, I.4 and I.5 (and by picking a smaller $\bar{\epsilon} > 0$ if needed), one can see the existence of some $c_0 > 0, \bar{\epsilon} > 0$ such that

$$\|\boldsymbol{x}_t(x)\| \leq e^{-c_0 t} \|x\| \quad \forall t \geq 0, \ x \in \overline{B(\boldsymbol{0}, \bar{\epsilon})}, \tag{J.3}$$

$$\|\nabla f(x)\| \geq c_0 \ \forall x \in \overline{\mathcal{G}} \backslash B(\boldsymbol{0}, \bar{\epsilon}), \tag{J.4}$$

$$\boldsymbol{d}\big(\boldsymbol{x}_t(x), \mathcal{G}^c\big) \geq \boldsymbol{d}(x, \mathcal{G}^c) + c_0 t \quad \text{if } \boldsymbol{x}_s(x) \in \mathcal{G} \text{ and } \boldsymbol{d}\big(\boldsymbol{x}_s(x), \mathcal{G}^c\big) \leq \bar{\epsilon} \ \forall s \in [0, t]. \tag{J.5}$$

Here $B(x, r) = \{y \in \mathbb{R}^d : \|x - y\| < r\}$ is the open ball centered at $x$ with radius $r > 0$. One immediate consequence from eq. (J.5) is that

$$\boldsymbol{d}\big(\boldsymbol{x}_t(x), \mathcal{G}^c\big) \geq \min\{\boldsymbol{d}(x, \mathcal{G}^c), \bar{\epsilon}\} \quad \forall x \in \mathcal{G}, t \geq 0. \tag{J.6}$$

Besides, the boundedness of domain $\mathcal{G}$ and smoothness of $f(\cdot)$ imply the existence of some $M > 0$ such that

$$\overline{\mathcal{G}} \subseteq B(\boldsymbol{0}, M), \tag{J.7}$$

$$\|\nabla f(x)\| \leq M \ \forall x \in \overline{\mathcal{G}}, \tag{J.8}$$

$$\big\|\nabla^2 f(x)\big\| \leq M \ \forall x \in \overline{\mathcal{G}} \tag{J.9}$$

where for the matrix norm we use spectral norm.

Recall the definitions of the configuration sets $\mathcal{A}(i_1, \cdots, i_m)$, the unique configuration $\boldsymbol{i}^*$ and count of jumps $k^*$ to trigger the exit with minimum cost (see Assumption I.7 and the remark underneath). The following three results provide control on the sizes and inter-arrival times of typical jumps that can trigger the escape from $\mathcal{G}$. The proofs involve analyses of the deterministic dynamical system $\widetilde{\boldsymbol{x}}_t$ and will be provided in Section K. In particular, we introduce another concept as *type* of jumps $(w_1, \cdots, w_k)$ that is similar configuration $(i_1, \cdots, i_m)$ and configuration sets $\mathcal{A}(i_1, \cdots, i_m)$. For any $k \in \mathbb{N}$ and any $(w_1, \cdots, w_k) \in (\mathbb{R}^d \backslash \mathbb{O})^k$, we say that the $(w_1, \cdots, w_k)$ is of type-$\boldsymbol{j}$ for some $\boldsymbol{j} = (j_1, \cdots, j_k) \in \{1, 2, \cdots, m\}^k$ iff

$$w_i \in E_{j_i} \ \forall i = 1, \cdots, k. \tag{J.10}$$

In other words, if $\boldsymbol{w}^1$ and $\boldsymbol{w}^2$ are of the same type $\boldsymbol{j}$, then they have the same cardinality $|\boldsymbol{w}^1| = |\boldsymbol{w}^2| = |\boldsymbol{j}| = k$ for some $k \in \mathbb{N}$; moreover, for any $i \in [k]$, the jumps $w_i^1$ and $w_i^2$ are both in the cone $E_{j_i}$ so both jumps point at directions in cone $E_{j_i}$ and have the same cost $J(w_i^1) = J(w_i^2) = \alpha_{j_i} - 1$ for some $j_i \in [m]$. Define the set

$$\mathcal{A}_{\boldsymbol{j}}^{\text{type}} = \{\boldsymbol{w} \in (\mathbb{R}^d \backslash \mathbb{O})^{|\boldsymbol{j}|} : \ \boldsymbol{w} \text{ is of type-}\boldsymbol{j}\}.$$

The accumulated cost for any $\boldsymbol{w} \in \mathcal{A}_{\boldsymbol{j}}^{\text{type}}$ is defined as $\mathcal{J}_{\text{type}}(\boldsymbol{j}) = \sum_{i=1}^{|\boldsymbol{j}|}(\alpha_{j_i} - 1)$. Lastly, define the skip-one accumulated cost for any type $\boldsymbol{j}$ as

$$\mathcal{J}_{\text{type}}^{\downarrow}(\boldsymbol{j}) = \max\{ \sum_{i=1, \cdots, |\boldsymbol{j}|, i \neq k} (\alpha_{j_i} - 1) : \ k = 1, 2, \cdots, |\boldsymbol{j}|\},$$

i.e., $\mathcal{J}_{\text{type}}^{\downarrow}(\boldsymbol{j})$ is the highest possible accumulated cost if we remove one element in $\boldsymbol{w}$ for $\boldsymbol{w} \in \mathcal{A}_{\boldsymbol{j}}^{\text{type}}$.

**Lemma J.1.** *There exist some $\bar{t} \in (0, \infty), \bar{\delta} \in (0, \infty), \epsilon_0 > 0$ such that the following claim holds. Let $k' \in \mathbb{N}$ and $\boldsymbol{j} = (j_1, \cdots, j_{k'}) \in \{1, 2, \cdots, m\}^{k'}$ be such that $\mathcal{J}_{type}^{\downarrow}(\boldsymbol{j}) < J_{\mathcal{G}}$. For any $\boldsymbol{t}^{(k')} = (t_1, \cdots, t_{k'}) \in \{0\} \times \mathbb{R}_+^{k'-1}, \boldsymbol{w}^{(k')} = (w_1, \cdots, w_{k'}) \in \mathcal{A}_{\boldsymbol{j}}^{type}$, the following set of conditions*

$$t_j < \bar{t} \ \forall j = 2, 3, \cdots, k' \tag{J.11}$$

$$\|w_j\| > \bar{\delta} \ \forall j = 1, 2, \cdots, k'. \tag{J.12}$$

*is the necessary condition for $\inf_{s \geq 0} \boldsymbol{d}\big(\widetilde{\boldsymbol{x}}(s, \mathbf{0}; \ \boldsymbol{t}^{(k)}, \boldsymbol{w}^{(k)}), \mathcal{G}^c\big) \leq \epsilon_0$.*

**Lemma J.2.** *Let $\bar{t}, \bar{\delta}, \epsilon_0$ be the positive constants prescribed in Lemma J.1. There exists some $\epsilon_1 > 0$ such that the following claim holds.*

*Let $k' \in \mathbb{N}$ and $\boldsymbol{j} = (j_1, \cdots, j_{k'}) \in \{1, 2, \cdots, m\}^{k'}$ be such that $\mathcal{J}_{type}^{\downarrow}(\boldsymbol{j}) < J_{\mathcal{G}}$. For any $x \in B(\mathbf{0}, \epsilon_1)$, any $\boldsymbol{t}^{(k')} = (t_1, \cdots, t_{k'}) \in \{0\} \times \mathbb{R}_+^{k'-1}, \boldsymbol{w}^{(k')} = (w_1, \cdots, w_{k'}) \in \mathcal{A}_{\boldsymbol{j}}^{type}$, the following set of conditions*

$$t_j < 2\bar{t} \ \forall j = 2, 3, \cdots, k' \tag{J.13}$$

$$\|w_j\| > \bar{\delta}/2 \ \forall j = 1, 2, \cdots, k'. \tag{J.14}$$

*is the necessary condition for $\inf_{s \geq 0} \boldsymbol{d}\big(\widetilde{\boldsymbol{x}}(s, x; \ \boldsymbol{t}^{(k)}, \boldsymbol{w}^{(k)}), \mathcal{G}^c\big) \leq \epsilon_0/2$.*

**Lemma J.3.** *There exist some $\epsilon_0 > 0, \delta_0 > 0$ such that*

$$\sup_{s \geq 0} \boldsymbol{d}\big(\widetilde{\boldsymbol{x}}(s, x; \ \boldsymbol{t}^{(k)}, \boldsymbol{w}^{(k)}), \mathcal{G}^c\big) > \epsilon_0$$

*for all $x \in B(\mathbf{0}, \delta_0)$, all $(i_1, \cdots, i_m) \in \mathbb{N}^m$ with $\mathcal{J}(i_1, \cdots, i_m) < J_{\mathcal{G}}$ and $k = \sum_{j=1}^m i_j$, and all $\boldsymbol{w}^{(k)} \in \mathcal{A}(i_1, \cdots, i_m), \boldsymbol{t}^{(k)} \in \{0\} \times \mathbb{R}_+^{k-1}$.*

Similar to the ODE path $\boldsymbol{x}$, we defined the following paths with rate $\eta > 0$:

$$(\dot{\boldsymbol{x}}_t^\eta)(x) = -\eta \nabla f\big(\boldsymbol{x}_t^\eta(x)\big) \ \forall t \geq 0, \tag{J.15}$$

$$\boldsymbol{x}_0^\eta(x) = x. \tag{J.16}$$

In other words, ODE flow $\boldsymbol{x}_t$ is equivalent to $\boldsymbol{x}_t^\eta$ with rate $\eta = 1$. The next result gives us an upper bound for the return time to a neighborhood of the local minimum as for the gradient flow.

**Lemma J.4.** *For $\tau_{ODE}^\eta(x, \epsilon) \triangleq \min\{t \geq 0 : \ \boldsymbol{x}_t^\eta(x) \in \overline{B(\mathbf{0}, \epsilon)}\}$, there exists $c_1 \in (0, \infty)$ such that for any $\epsilon \in (0, \bar{\epsilon})$,*

$$\tau_{ODE}^\eta(x, \epsilon) \leq \frac{c_1 + c_1 \log(1/\epsilon)}{\eta} \ \ \forall x \in \overline{\mathcal{G}}.$$

*Proof.* Due to eq. (J.4), we know that $T_0 \triangleq \frac{\sup_{x \in \mathcal{G}} f(x) - \inf_{x \in \mathcal{G}} f(x)}{c_0} < \infty$ and

$$\sup_{x \in \overline{\mathcal{G}}} \tau_{ODE}^1(x, \bar{\epsilon}) \leq T_0.$$

Furthermore, from eq. (J.3) one can see that

$$\sup_{x \in \overline{\mathcal{G}}} \tau_{ODE}^1(x, \epsilon) - \tau_{ODE}^1(x, \bar{\epsilon}) \leq \frac{\log(1/\epsilon) - \log(1/\bar{\epsilon})}{c_0}.$$

Now combining these two bounds with a $1/\eta$ time scaling, we have

$$\sup_{x \in \overline{\mathcal{G}}} \tau_{ODE}^\eta(x, \epsilon) \leq \frac{c_1 + c_1 \log(1/\epsilon)}{\eta}$$

for $c_1 = \max\{T_0 + \frac{\log(\bar{\epsilon})}{c_0}, \frac{1}{c_0}\}$. $\qquad\square$

With Lemma J.1-J.4, we are able to pick some constants to facilitate the analysis below. By decreasing $\bar{\epsilon}$ and increasing $M$ if needed, we can assume the existence of some $\bar{t} < \infty, \bar{\delta} < \bar{\epsilon}$ such that

- (Due to Lemma J.1 and J.2) For any $k \in \mathbb{N}$ and $\boldsymbol{j} \in \{1, 2, \cdots, m\}^k$ such that $\mathcal{J}_{\text{type}}^{\downarrow}(\boldsymbol{j}) < J_{\mathcal{G}}$, any $x \in B(\mathbf{0}, 100l^*\bar{\epsilon})$, any $\boldsymbol{t}^{(k)} = (t_1, \cdots, t_k) \in \{0\} \times \mathbb{R}_+^{k-1}$, $\boldsymbol{w}^{(k)} = (w_1, \cdots, w_k) \in \mathcal{A}_{\boldsymbol{j}}^{\text{type}}$ with

$$\boldsymbol{d}\big(\widetilde{\boldsymbol{x}}(\sum_{j=1}^{k} t_j, x; \ \boldsymbol{t}^{(k)}, \boldsymbol{w}^{(k)}), \mathcal{G}^c\big) < 100l^*\bar{\epsilon},$$

  it holds that

$$t_j < \bar{t} \ \forall j = 2, 3, \cdots, k \text{ and } \|w_j\| > \bar{\delta} \ \forall j = 1, 2, \cdots, k. \tag{J.17}$$

- For any $(i_1, \ldots, i_m) \in \mathbb{N}^m$ with $\mathcal{J}(i_1, \ldots, i_m) < J_{\mathcal{G}}$,

$$\boldsymbol{d}\big(\mathcal{G}(i_1, \cdots, i_m), \mathcal{G}^c\big) > 100l^*\bar{\epsilon}; \tag{J.18}$$

- There exists some $x \in \mathcal{G}(\boldsymbol{i}^*)$ such that

$$\boldsymbol{d}(x, \mathcal{G}) > 100l^*\bar{\epsilon}; \tag{J.19}$$

- (Due to Lemma J.3) The constant $\bar{\epsilon} > 0$ is sufficiently small such that

$$\sup_{s \geq 0} \boldsymbol{d}\big(\widetilde{\boldsymbol{x}}(s, x; \ \boldsymbol{t}^{(k)}, \boldsymbol{w}^{(k)}), \mathcal{G}^c\big) > 100l^*\bar{\epsilon} \tag{J.20}$$

  for all $x \in B(\mathbf{0}, 100l^*\bar{\epsilon})$, all $(i_1, \cdots, i_m) \in \mathbb{N}^m$ with $\mathcal{J}(i_1, \cdots, i_m) < J_{\mathcal{G}}$ and $k = \sum_{j=1}^{m} i_j$, and all $\boldsymbol{w}^{(k)} \in \mathcal{A}(i_1, \cdots, i_m)$, $\boldsymbol{t}^{(k)} \in \{0\} \times \mathbb{R}_+^{k-1}$;

- (Due to Lemma J.4) The constant $\bar{t} < \infty$ is large enough so that

$$\sup_{x \in \bar{\mathcal{G}}} \tau_{\text{ODE}}^{\eta}(x, \bar{\epsilon}) < \bar{t}/\eta.$$

## J.2 BOUNDING THE DISTANCE BETWEEN ODE FLOW AND SGD ITERATES

Moving on, we show that, without large jumps, the SGD iterates $X_n^{\eta}(x)$ are unlikely to show significant deviation from the deterministic gradient descent process $\mathbf{y}_n^{\eta}(x)$ defined as

$$\mathbf{y}_0^{\eta}(x) = x, \tag{J.21}$$

$$\mathbf{y}_n^{\eta}(x) = \mathbf{y}_{n-1}^{\eta}(x) - \eta \nabla f\big(\mathbf{y}_{n-1}^{\eta}(x)\big). \tag{J.22}$$

We are ready to state the first lemma, where we bound the distance between the gradient descent iterates $\mathbf{y}_n^{\eta}(y)$ and the ODE $\mathbf{x}^{\eta}(t, x)$ when the initial conditions $x, y$ are close enough.

**Lemma J.5.** *Given $t > 0$ and $x, y \in \mathcal{G}$ such that the line segment between $\boldsymbol{x}^{\eta}(k, x)$ and $\boldsymbol{y}_k^{\eta}(y)$ lies in $\mathcal{G}$ for any $k = 1, 2, \cdots, \lfloor t \rfloor$,*

$$\sup_{s \in [0,t]} \left\| \boldsymbol{x}^{\eta}(s, x) - \boldsymbol{y}_{\lfloor s \rfloor}^{\eta}(y) \right\| \leq (2\eta M + \|x - y\|) \exp(\eta M t)$$

*where $M \in (0, \infty)$ is the constant in eq. (J.8)eq. (J.9).*

*Proof.* Define a continuous-time process $\mathbf{y}^{\eta}(s; y) \triangleq \mathbf{y}_{\lfloor s \rfloor}^{\eta}(y)$, and note that

$$\mathbf{x}^{\eta}(s, x) = \mathbf{x}^{\eta}(\lfloor s \rfloor, x) - \eta \int_{\lfloor s \rfloor}^{s} \nabla f(\mathbf{x}^{\eta}(u, x)) du$$

$$\mathbf{x}^{\eta}(\lfloor s \rfloor, x) = x - \eta \int_{0}^{\lfloor s \rfloor} \nabla f(\mathbf{x}^{\eta}(u, x)) du$$

$$\mathbf{y}_{\lfloor s \rfloor}^{\eta}(y) = \mathbf{y}^{\eta}(\lfloor s \rfloor, y) = y - \eta \int_{0}^{\lfloor s \rfloor} \nabla f(\mathbf{y}^{\eta}(u, y)) du.$$

Therefore, if we define function

$$b(u) = \mathbf{x}^{\eta}(u, x) - \mathbf{y}^{\eta}(u, y),$$

from the fact $\|\nabla f(\cdot)\| \leq M$, one can see that $\|b(u)\| \leq \eta M + \|x - y\|$ for any $u \in [0, 1)$ and $\|b(1)\| \leq 2\eta M + \|x - y\|$. In case that $s > 1$, from the display above and the fact $\|\nabla^2 f(\cdot)\| \leq M$, we now have

$$\left\| \mathbf{y}_{\lfloor s \rfloor}^{\eta}(x) - \mathbf{x}^{\eta}(s, x) \right\| \leq \|b(\lfloor s \rfloor)\| + \eta M;$$

$$\|b(\lfloor s \rfloor)\| \leq \eta M \int_1^{\lfloor s \rfloor} \|b(u)\| \, du.$$

From Gronwall's inequality (see Theorem 68, Chapter V of Protter (2005), where we let function $\alpha(u)$ be $\alpha(u) = \|b(u + 1)\|$), we have

$$\left\| \mathbf{y}_{\lfloor s \rfloor}^{\eta}(x) - \mathbf{x}^{\eta}(s, x) \right\| \leq (2\eta M + \|x - y\|) \exp(\eta M t).$$

This concludes the proof. $\qquad\square$

Now we consider an extension of the previous Lemma in the following sense: when both perturbed by at most $l^*$ similar perturbations, the ODE flow and gradient descent process should still stay close enough. Analogous to the definition of the perturbed ODE $\widetilde{\boldsymbol{x}}^{\eta}$, we can construct a process $\widetilde{\boldsymbol{y}}^{\eta}$ as a perturbed gradient descent process as follows. For some integer $1 \leq l \leq l^*$, a sequence of strictly positive integers $(t_2, \cdots, t_l)$ (with convention $t_1 = 0$ and let $\boldsymbol{t}^{(l)} = (t_1, \cdots, t_l)$), a sequence of non-zero vectors $\widetilde{\boldsymbol{w}}^{(l)} = (\widetilde{w}_1, \cdots, \widetilde{w}_l)$ and $y \in \mathbb{R}^d$, define the perturbed gradient descent iterates with gradient clipping at $b$ as

$$\widetilde{\boldsymbol{y}}_n^{\eta}(y; \boldsymbol{t}^{(l)}, \widetilde{\boldsymbol{w}}^{(l)})$$

$$= \widetilde{\boldsymbol{y}}_{n-1}^{\eta}(y; \boldsymbol{t}^{(l)}, \widetilde{\boldsymbol{w}}^{(l)}) + \varphi_b\big( -\eta \nabla f(\widetilde{\boldsymbol{y}}_{n-1}^{\eta}(y; \boldsymbol{t}^{(l)}, \widetilde{\boldsymbol{w}}^{(l)})) + \sum_{j=2}^l \mathbb{1}\{n = t_1 + \cdots + t_j\} \widetilde{w}_j \quad \text{(J.23)}$$

with initial condition $\widetilde{\boldsymbol{y}}_0^{\eta}(y; \boldsymbol{t}^{(l)}, \widetilde{\boldsymbol{w}}^{(l)}) = y + \varphi_b(\widetilde{w}_1)$.

**Corollary J.6.** *For any fixed $\epsilon > 0, t > 0$, the following claim holds for all $\eta > 0$ sufficiently small: given any $x, y \in \mathcal{G}$, some integer $1 \leq l \leq l^*$, a sequence of strictly positive integers $\boldsymbol{t}^{(l)} = (t_j)_{j=2}^l$ (with convention $t_1 = 0$), and any two sequences of vectors $\boldsymbol{w}^{(l)} = (w_j)_{j=1}^l, \widetilde{\boldsymbol{w}}^{(l)} = (\widetilde{w}_j)_{j \geq 1}$ such that (let $t_{end} = \sum_{j=1}^l t_j$)*

- *$|x - y| < \epsilon$;*
- *$t_j \leq 2t/\eta$ for all $j \in [l]$;*
- *$|w_j - \widetilde{w}_j| < \epsilon$ for all $j \in [l]$;*
- *The line segment between $\widetilde{\boldsymbol{x}}^{\eta}(k, x; \boldsymbol{t}^{(l)}, \boldsymbol{w}^{(l)})$ and $\widetilde{\boldsymbol{y}}_k^{\eta}(y; \boldsymbol{t}^{(l)}, \widetilde{\boldsymbol{w}}^{(l)})$ lies in $\mathcal{G}$ for any $k \leq t_{end} - 1$,*

*then we have*

$$\sup_{s \in [0, t_{end}]} \left\| \widetilde{\boldsymbol{x}}^{\eta}(s, x; \boldsymbol{t}^{(l)}, \boldsymbol{w}^{(l)}) - \widetilde{\boldsymbol{y}}_{\lfloor s \rfloor}^{\eta}(y; \boldsymbol{t}^{(l)}, \widetilde{\boldsymbol{w}}^{(l)}) \right\| \leq \bar{\rho}(t)\epsilon$$

*where the constant $\bar{\rho} = (2 \exp(2Mt) + 2)^{l^*}$. In particular, $\bar{\rho} = \bar{\rho}(\bar{t})$ is a constant where $\bar{t}$ is the constant in eq. (J.11).*

*Proof.* Throughout this proof, fix some $\eta \in (0, \epsilon/2M)$. We will show that for any $\eta$ in the range the claim would hold.

First, on interval $[0, t_2)$, from Lemma J.5, one can see that (since $2M\eta < \epsilon$)

$$\sup_{s \in [0, t_2)} \left\| \widetilde{\boldsymbol{x}}^{\eta}(s, x; \boldsymbol{s}^{(l)}, \boldsymbol{w}^{(l)}) - \widetilde{\boldsymbol{y}}_{\lfloor s \rfloor}^{\eta}(y; \boldsymbol{t}^{(l)}, \widetilde{\boldsymbol{w}}^{(l)}) \right\| \leq 2 \exp(2Mt) \cdot \epsilon.$$

The at $t = t_2$, by considering the difference between $w_2$ and $\widetilde{w}_2$, and the possible change due to one more gradient descent step (which is bounded by $\eta M < \epsilon$), we have

$$\sup_{s \in [0,t_2]} \left\| \widetilde{\boldsymbol{x}}^\eta(s, x; \boldsymbol{s}^{(l)}, \boldsymbol{w}^{(l)}) - \widetilde{\boldsymbol{y}}^\eta_{\lfloor s \rfloor}(y; \boldsymbol{t}^{(l)}, \widetilde{\boldsymbol{w}}^{(l)}) \right\| \leq (2 \exp(2Mt) + 2) \cdot \epsilon.$$

Now we proceed inductively. For any $j = 2, 3, \cdots, l - 1$, assume that

$$\sup_{s \in [0,t_j]} \left\| \widetilde{\boldsymbol{x}}^\eta(s, x; \boldsymbol{s}^{(l)}, \boldsymbol{w}^{(l)}) - \widetilde{\boldsymbol{y}}^\eta_{\lfloor s \rfloor}(y; \boldsymbol{t}^{(l)}, \widetilde{\boldsymbol{w}}^{(l)}) \right\| \leq (2 \exp(2Mt) + 2)^{j-1} \cdot \epsilon.$$

Then by focusing on interval $[t_j, t_{j+1}]$ and using Lemma J.5 again, one can show that

$$\sup_{t \in [t_j, t_{j+1}]} \left\| \widetilde{\boldsymbol{x}}^\eta(s, x; \boldsymbol{s}^{(l)}, \boldsymbol{w}^{(l)}) - \widetilde{\boldsymbol{y}}^\eta_{\lfloor s \rfloor}(y; \boldsymbol{t}^{(l)}, \widetilde{\boldsymbol{w}}^{(l)}) \right\| \leq 2\epsilon + \left( (2 \exp(2Mt) + 2)^{j-1} + 1 \right) \exp(2Mt)\epsilon$$

$$\leq (2 \exp(2Mt) + 2)^j \cdot \epsilon.$$

This concludes the proof. $\qquad\square$

Using an almost identical approach, one can show the following results that bounds the distance between two ODE flows with similar initial values and jumps.

**Corollary J.7.** *There exists some constant $\rho^* \in (0, \infty)$ such that the following claim holds: For any fixed $\epsilon > 0$, any $x, y \in \mathcal{G}$, any integer $1 \leq l \leq l^*$, a sequence of strictly positive integers $\boldsymbol{t}^{(l)} = (t_j)_{j=2}^l$ (with convention $t_1 = 0$), and any two sequences of vectors $\boldsymbol{w}^{(l)} = (w_j)_{j=1}^l$, $\widetilde{\boldsymbol{w}}^{(l)} = (\widetilde{w}_j)_{j \geq 1}^l$ such that (let $t_{end} = \sum_{j=1}^l t_j$)*

- $|x - y| < \epsilon$;
- $t_j \leq 2\bar{t}/\eta$ *for all* $j \in [l]$;
- $|w_j - \widetilde{w}_j| < \epsilon$ *for all* $j \in [l]$;
- *The line segment between $\widetilde{\boldsymbol{x}}^\eta(s, x; \boldsymbol{t}^{(l)}, \boldsymbol{w}^{(l)})$ and $\widetilde{\boldsymbol{x}}^\eta(s, y; \boldsymbol{t}^{(l)}, \widetilde{\boldsymbol{w}}^{(l)})$ lies in $\mathcal{G}$ for any $s < t_{end}$,*

*then we have*

$$\sup_{s \in [0, t_{end}]} \left\| \widetilde{\boldsymbol{x}}^\eta(s, x; \boldsymbol{t}^{(l)}, \boldsymbol{w}^{(l)}) - \widetilde{\boldsymbol{x}}^\eta(s, y; \boldsymbol{t}^{(l)}, \widetilde{\boldsymbol{w}}^{(l)}) \right\| \leq \rho^* \epsilon.$$

In the next few results, we show that a similar type of control can be obtained on the distance between gradient descent iterates $\widetilde{\boldsymbol{y}}^\eta_n$ and the SGD iterates $X^\eta_n$. Specifically, our first goal is to show that before any *large* jump, it is unlikely that the (deterministic) gradient descent process $\boldsymbol{y}^\eta_n$ would deviate too far from $X^\eta_n$. To facilitate the analysis below, we introduce some additional notations. First, we will group the noises $Z_n$ based on a threshold level $\delta > 0$: let us define

$$Z^{\leq \delta, \eta}_n \triangleq Z_n \mathbb{1}\{\eta \|Z_n\| \leq \delta\}, \tag{J.24}$$

$$Z^{> \delta, \eta}_n \triangleq Z_n \mathbb{1}\{\eta \|Z_n\| > \delta\}. \tag{J.25}$$

The former are viewed as *small noises* while the latter will be referred to as *large noises* or *large jumps*. Furthermore, for any $j \geq 1$, define the $j^{\text{th}}$ arrival time, size, and direction of large jumps as

$$T^\eta_j(\delta) \triangleq \min\{n > T^\eta_{j-1}(\delta) : \eta \|Z_n\| > \delta\}, \quad T^\eta_0(\delta) = 0, \tag{J.26}$$

$$W^\eta_j(\delta) \triangleq Z_{T^\eta_j(\delta)}, \tag{J.27}$$

$$\Theta^\eta_j(\delta) \triangleq W^\eta_j(\delta) / \left\| W^\eta_j(\delta) \right\|. \tag{J.28}$$

The following event

$$A(n, \eta, \epsilon, \delta) = \left\{ \max_{k=1,2,\cdots,n \wedge (T^\eta_1(\delta)-1)} \eta \|Z_1 + \cdots + Z_k\| \leq \epsilon \right\} \tag{J.29}$$

describes a scenario where, before the first large jump, not much perturbation has been caused by noises.

**Lemma J.8.** *For any $\epsilon > 0, \eta > 0$, any $x, y \in \mathcal{G}$ and positive integer $n$ such that $\|x - y\| < \frac{\epsilon}{2 \exp(\eta M n)}$, on event*

$$A(n, \eta, \frac{\epsilon}{2 \exp(\eta M n)}, \delta) \bigcap$$

$$\left\{ \text{the line segment between } \boldsymbol{y}_j^\eta(y) \text{ and } X_j^\eta(x) \text{ lies in } \mathcal{G} \ \forall j = 0, 1, \cdots, n \wedge (T_1^\eta(\delta) - 1) \right\},$$

*we have*

$$\|\boldsymbol{y}_m^\eta(y) - X_m^\eta(x)\| < \epsilon \ \forall m = 1, 2, \cdots, n \wedge (T_1^\eta(\delta) - 1).$$

*Proof.* On the said event, we are able to apply Lemma G.5 and obtain that

$$\left\|\boldsymbol{y}_j^\eta(y) - X_j^\eta(x)\right\| \leq (\|x - y\| + \frac{\epsilon}{2 \exp(\eta M n)}) \exp(\eta M j) < \epsilon \ \ \forall j = 1, 2, \cdots, n \wedge (T_1^\eta(\delta) - 1)$$

and conclude the proof. $\qquad\square$

Similar to the extension from Lemma J.5 to Corollary J.6, we can extend Lemma J.8 to show that, if we consider the a gradient descent process that is only perturbed by large noises, then it should stay pretty close to the SGD iterates $X_n^\eta$. To be specific, let

$$Y_0^\eta(x) = x$$

$$Y_n^\eta(x) = Y_{n-1}^\eta(x) - \varphi_b\big( -\eta \nabla f\big(Y_{n-1}^\eta(x)\big) + \sum_{j \geq 1} \mathbb{1}\{n = T_j^\eta(\delta)\} \eta Z_n\big).$$

be a gradient descent process (with gradient clipping at threshold $b$) that is only perturbed by large noises. The next corollary can be shown by an approach identical to the one for Corollary J.6 (namely, inductively repeating Lemma J.8 at arrival time of each jump) so we omit the details here.

**Corollary J.9.** *There exists some function $\widetilde{\rho} : (0, \infty) \mapsto (0, \infty)$ such that the following claim hold for all $\epsilon > 0, t > 0$ and all sufficiently small $\eta > 0$: For any $x, y \in \mathcal{G}$) with $\|x - y\| < \epsilon$ and any $l \in [l^*]$, on event $A_0(\epsilon, \eta, \delta, l, t) \cap B_0(\epsilon, \eta, \delta, l, t) \cap C_0(\epsilon, \eta, \delta, l, t)$, we have*

$$\|Y_n^\eta(y) - X_n^\eta(x)\| < \widetilde{\rho}(t)\epsilon \ \forall n = 1, 2, \cdots, T_l^\eta(\delta)$$

*where*

$$A_0(\epsilon, \eta, \delta, l, t) \triangleq \left\{ \forall i = 1, \cdots, l, \max_{j = T_{i-1}^\eta(\delta)+1, \cdots, T_i^\eta(\delta)-1} \eta \left\| Z_{T_{i-1}^\eta(\delta)+1} + \cdots + Z_j \right\| \leq \frac{\epsilon}{2 \exp(2tM)} \right\};$$

$$B_0(\epsilon, \eta, \delta, l, t) \triangleq \left\{ \forall j = 2, \cdots, l, T_j^\eta(\delta) - T_{j-1}^\eta(\delta) \leq 2t/\eta \right\},$$

$$C_0(\epsilon, \eta, \delta, l, t) \triangleq \{\text{the line segment between } Y_j^\eta(y) \text{ and } X_j^\eta(x) \text{ lies in } \mathcal{G} \ \forall j = 0, 1, \cdots, T_l^\eta(\delta) - 1\}.$$

*In particular, $\widetilde{\rho} = \widetilde{\rho}(\bar{t})$ is a constant*

The next result shows that the type of events $A(n, \eta, \epsilon, \delta)$ defined in eq. (J.29) is indeed very likely to occur, especially for small $\epsilon$. For clarity of the presentation, we introduce the following definitions that are slightly more general than the *small* and *large* jumps defined in eq. (J.24)eq. (J.25) (for any $c > 0$)

$$Z_n^{\leq c} \triangleq Z_n \mathbb{1}\{\|Z_n\| \leq c\},$$

$$Z_n^{>c} \triangleq Z_n \mathbb{1}\{\|Z_n\| > c\}.$$

**Lemma J.10.** *Given any $t > 0$, $\epsilon > 0$, and $N > 0$, the following holds for any sufficiently small $\delta > 0$:*

$$\mathbb{P}\Big( \max_{n=1,2,\cdots,\lceil t/\eta \rceil} \eta \left\| Z_1^{\leq \delta/\eta} + \cdots + Z_n^{\leq \delta/\eta} \right\| > \epsilon \Big) = o(\eta^N)$$

*as $\eta \downarrow 0$.*

*Proof.* For any $C \in (0, \infty), \delta > 0, \beta \in (0,1)$, consider the following decomposition of the small noise $Z_n^{\leq \delta/\eta}$. Here we adopt the convention that $E_0 = \mathbb{R}^d \backslash (\cup_{j=1}^m E_j)$. (For definitions and properties of the closed cones $E_1, \cdots, E_m$, see Assumption I.6 and the remark underneath.)

$$Z_n^{(j)} \triangleq Z_n \mathbb{1}\{Z_n \in E_j\},$$

$$Z_n^{\leq C, (j)} \triangleq Z_n^{(j)} \mathbb{1}\{\left\| Z_n^{(j)} \right\| \leq C\},$$

$$Z_{C,\delta,\eta}^{\downarrow,(j)}(n) \triangleq Z_n^{(j)} \mathbb{1}\{\left\| Z_n^{(j)} \right\| \in (C, \delta/\eta]\}$$

Based on this decomposition, we define the following iid random variable sequences

$$\widetilde{z}^{(j)}(n) \triangleq Z_n^{\leq C, (j)} - \mathbb{E} Z_n^{\leq C, (j)},$$

$$\widetilde{Z}^{(j)}(n) \triangleq \left\| Z_{C,\delta,\eta}^{\downarrow,(j)}(n) \right\|,$$

$$\widetilde{Z}_\beta^{(j)}(n) \triangleq \left\| Z_n^{\leq 1/\eta^\beta, (j)} \right\|.$$

Meanwhile, define the projection operators $\mathbf{P}_i : \mathbb{R}^d \mapsto \mathbb{R}$ with $\mathbf{P}_i(v_1, \cdots, v_d) = v_i$. First, recall that $\mathbb{E} Z_1 = \mathbf{0}$. So for all $C$ sufficiently large, we have

$$\sup_{y \geq C} \left\| \mathbb{E} Z_1^{\leq y} \right\| = \sup_{y \geq C} \left\| \mathbb{E} Z_1^{>y} \right\| < \frac{\epsilon}{2t}.$$

Moreover, due to $H_j \in RV_{-(\alpha_j+1)}$ for all $j = 1, 2, \cdots, m$ and $H_0(x) = o(1/x^\alpha)$ for any $\alpha > 0$ (for definition of functions $H_j$, see eq. (I.5)), we know that $\mathbb{E} \left\| Z_n^{(j)} \right\| < \infty$ for any $j = 0, 1, \cdots, m$, implying that for all $C$ large enough, we have (for any $j = 0, 1, \cdots, m$)

$$\sup_{y,z \geq C} \left\| \mathbb{E} Z_1^{\leq y, (j)} - \mathbb{E} Z_1^{\leq z, (j)} \right\| < \frac{1}{2t} \cdot \frac{\epsilon}{24\sqrt{m+1}}. \tag{J.30}$$

Fix such $C$. Besides, recall that in Assumption I.6 we have $1 < \alpha_1 < \alpha_2 < \cdot < \alpha_m$. Therefore, we are able fix some $\beta \in (0,1)$ and $\delta > 0$ so that

$$\beta \alpha_1 > 1, \tag{J.31}$$

$$(N+1)\delta < \frac{\epsilon}{12\sqrt{m+1}}, \tag{J.32}$$

$$\left( \frac{N}{\beta \alpha_1 - 1} + 1 \right) \delta < \frac{\epsilon}{12\sqrt{m+1}}. \tag{J.33}$$

With the fixed $C$ and $\delta$, note that eventually $\delta/\eta > C$ as $\eta \downarrow 0$. Therefore, for all $\eta$ sufficiently small,

$$\mathbb{P}\Big( \max_{n=1,2,\cdots,\lceil t/\eta \rceil} \eta \left\| Z_1^{\leq \delta/\eta} + \cdots + Z_n^{\leq \delta/\eta} \right\| > \epsilon \Big)$$

$$\leq \mathbb{P}\Big( \max_{n=1,2,\cdots,\lceil t/\eta \rceil} \eta \left\| \sum_{k=1}^n Z_k^{\leq \delta/\eta} - \mathbb{E} Z_k^{\leq \delta/\eta} \right\| > \frac{\epsilon}{2} \Big)$$

$$= \mathbb{P}\Big( \max_{n=1,2,\cdots,\lceil t/\eta \rceil} \eta \left\| \sum_{j=0}^m \sum_{k=1}^n Z_k^{\leq C, (j)} - \mathbb{E} Z_k^{\leq C, (j)} + Z_{C,\delta,\eta}^{\downarrow,(j)}(k) + \mathbb{E} Z_k^{\leq C, (j)} - \mathbb{E} Z_k^{\leq \delta/\eta, (j)} \right\| > \frac{\epsilon}{2} \Big)$$

$$\leq \sum_{j=0}^m \mathbb{P}\Big( \max_{n=1,2,\cdots,\lceil t/\eta \rceil} \eta \left\| \sum_{k=1}^n Z_k^{\leq C, (j)} - \mathbb{E} Z_k^{\leq C, (j)} + Z_{C,\delta,\eta}^{\downarrow,(j)}(k) + \mathbb{E} Z_k^{\leq C, (j)} - \mathbb{E} Z_k^{\leq \delta/\eta, (j)} \right\|$$

$$> \frac{\epsilon}{2\sqrt{m+1}} \Big)$$

$$\leq \sum_{j=0}^m \mathbb{P}\Big( \max_{n=1,2,\cdots,\lceil t/\eta \rceil} \eta \left\| \sum_{k=1}^n \widetilde{z}^{(j)}(k) \right\| > \frac{\epsilon}{6\sqrt{m+1}} \Big) \tag{J.34}$$

$$+ \sum_{j=0}^m \mathbb{P}\Big( \eta \sum_{k=1}^{\lceil t/\eta \rceil} \widetilde{Z}^{(j)}(k) > \frac{\epsilon}{6\sqrt{m+1}} \Big). \tag{J.35}$$

We bound the two terms eq. (J.34) and eq. (J.35) respectively. First, observe that

$$eq. \text{ (J.34)} \leq \sum_{j=0}^{m} \sum_{i=1}^{d} \mathbb{P}\Big( \max_{n=1,2,\cdots,\lceil t/\eta \rceil} \eta \big| \sum_{k=1}^{n} \mathbf{P}_i\big( \widetilde{z}^{(j)}(k) \big) \big| > \frac{\epsilon}{6\sqrt{(m+1)d}} \Big).$$

Let $\widetilde{z}_i^{(j)}(k) = \mathbf{P}_i\big( \widetilde{z}^{(j)}(k) \big)$ and let $\widetilde{z}_i^{(j)}$ be an iid copy. By definition, $|\widetilde{z}_i^{(j)}| \leq 2C$. Then from Hoeffding's inequality,

$$eq. \text{ (J.34)} \leq 2(m+1)d \cdot \exp\Big( -\frac{1}{2} \cdot \big( \frac{\epsilon}{6\sqrt{d(m+1)}} \big)^2 \cdot \frac{1/\eta^2}{2C\lceil t/\eta \rceil} \Big) = o(\eta^N).$$

As for term eq. (J.35), w.l.o.g. let us fix some $j = 0, 1, \cdots, m$ and bound

$$\mathbb{P}\Big( \eta \sum_{n=1}^{\lceil t/\eta \rceil} \widetilde{Z}^{(j)}(n) > \frac{\epsilon}{6\sqrt{m+1}} \Big) \tag{J.36}$$

$$= \sum_{i=0}^{k-1} \underbrace{\mathbb{P}\Big( \eta \sum_{n=1}^{\lceil t/\eta \rceil} \widetilde{Z}^{(j)}(n) > \frac{\epsilon}{6\sqrt{m+1}}, \ I = i \Big)}_{\triangleq \text{(I)}} + \underbrace{\mathbb{P}\Big( \eta \sum_{n=1}^{\lceil t/\eta \rceil} \widetilde{Z}^{(j)}(n) > \frac{\epsilon}{6\sqrt{m+1}}, \ I \geq k \Big)}_{\triangleq \text{(II)}}$$

where $I \triangleq \#\{n = 1, 2, \cdots, \lceil t/\eta \rceil : \ \widetilde{Z}^{(j)}(n) > 1/\eta^\beta\}$ and $k$ is some positive integer such that

$$k\delta < \frac{\epsilon}{12\sqrt{m+1}}, \quad k(\beta\alpha_1 - 1) > N.$$

Note that we can find such $k$ due to our choice of constants in eq. (J.31)-eq. (J.33).

For (I) and any $i = 0, 1, \cdots, k-1$, observe that (the non-negative RVs $\widetilde{Z}_\beta(n)$ and $\widetilde{Z}(n)$ are defined at the beginning of the proof)

$$\text{(I)} \leq \binom{\lceil t/\eta \rceil}{i} \cdot \mathbb{P}\Big( \eta \sum_{n=1}^{\lceil t/\eta \rceil - i} \widetilde{Z}^{(j)}(n) > \frac{\epsilon}{6\sqrt{m+1}} - i\eta \cdot \frac{\delta}{\eta}, \ \widetilde{Z}^{(j)}(n) \leq 1/\eta^\beta \ \forall n \in [\lceil t/\eta \rceil - i] \Big)$$

$$\leq \lceil t/\eta \rceil^i \cdot \mathbb{P}\Big( \sum_{n=1}^{\lceil t/\eta \rceil - i} \widetilde{Z}^{(j)}(n) > \frac{\epsilon}{12\sqrt{m+1}} \cdot \frac{1}{\eta}, \ \widetilde{Z}^{(j)}(n) \leq 1/\eta^\beta \ \forall n \in [\lceil t/\eta \rceil - i] \Big)$$

$$\leq \lceil t/\eta \rceil^i \cdot \mathbb{P}\Big( \sum_{n=1}^{\lceil t/\eta \rceil} \widetilde{Z}_\beta^{(j)}(n) > \frac{\epsilon}{12\sqrt{m+1}} \cdot \frac{1}{\eta} \Big).$$

Let $\widetilde{W}^{(j)}(n) \triangleq \widetilde{Z}_\beta^{(j)}(n) - \mathbb{E}\widetilde{Z}_\beta^{(j)}(n)$ and let $\widetilde{W}^{(j)}$ be an iid copy. Due to eq. (J.30), for all $\eta$ sufficiently small we will have that $\lceil t/\eta \rceil |\mathbb{E}\widetilde{Z}_\beta^{(j)}(1)| < \frac{\epsilon}{24\sqrt{m+1}} \cdot \frac{1}{\eta}$. Therefore, for all $\eta$ sufficiently small,

$$\text{(I)} \leq \lceil t/\eta \rceil^i \cdot \mathbb{P}\Big( \big| \sum_{n=1}^{\lceil t/\eta \rceil} \widetilde{W}^{(j)}(n) \big| > \frac{\epsilon}{24\sqrt{m+1}} \cdot \frac{1}{\eta} \Big). \tag{J.37}$$

Observe that $|\widetilde{W}^{(j)}| \leq 2/\eta^\beta$ and

$$\mathbb{E}(\widetilde{W}^{(j)})^2 = var\widetilde{Z}_\beta^{(j)} \leq \mathbb{E}(\widetilde{Z}_\beta^{(j)})^2 = \int_0^\infty 2y\mathbb{P}(\widetilde{Z}_\beta^{(j)} > y)dy = \int_0^{1/\eta^\beta} 2yH_j(y)dy.$$

Note that there are only two possibilities:

- If $j \neq 0$ and $\alpha_j \leq 2$, then from Karamata's theorem, we have $\mathbb{E}(\widetilde{Z}_\beta^{(j)})^2 \in RV_{-\beta(2-\alpha_j)}(\eta)$ and note that $\beta(2 - \alpha_j) < 1$;

- Otherwise, one can find some $\Delta > 0$ such that $H_j(x) = o(1/x^{2+\Delta})$, implying that $\mathbb{E}(\widetilde{Z}_\beta^{(j)})^2 \leq \int_0^\infty 2yH_j(y)dy < \infty$ regardless of the actual value of $\eta > 0$.

In any case, using Bernstein's inequality, we can conclude that

$$eq. \text{ (J.37)} \leq \lceil t/\eta \rceil^i \cdot 2 \exp \Big( - \frac{\frac{1}{2}\big(\frac{\epsilon}{12\sqrt{m+1}}\big)^2/\eta^2}{\lceil t/\eta \rceil \mathbb{E}(\widetilde{W}^{(j)})^2 + \frac{1}{3}\big(\frac{\epsilon}{12\sqrt{m+1}}\big)/\eta^{1-\beta}} \Big) = o(\eta^N).$$

On the other hand,

$$\text{(II)} \leq \mathbb{P}(I \geq k) \leq \binom{\lceil t/\eta \rceil}{k} \cdot \mathbb{P}\Big( \widetilde{Z}^{(j)}(n) > 1/\eta^\beta \; \forall n = 1, \dots, k \Big) \leq \lceil t/\eta \rceil^k \cdot \Big( H_j(1/\eta^\beta) \Big)^k,$$

which is upper bounded by a function regularly varying w.r.t. $\eta$ with index $k(\beta\alpha_1 - 1) > N$. Therefore, (II) $= o(\eta^N)$ and we conclude the proof. $\qquad\square$

## J.3 ANALYZING DIFFERENT SCENARIOS BEFORE THE FIRST EXIT OR RETURN

Using results and arguments above, we are able to illustrate the typical behavior of the SGD iterates $X_n^\eta$ in the following two scenarios. First, we show that, when starting from anywhere in $\mathcal{G}$, the SGD iterates $X_n^\eta$ will most likely return to the neighborhood of the $\mathbf{0}$ within a short period of time without exiting $\Omega$.

**Lemma J.11.** *For any $\epsilon \in (0, \bar{\epsilon})$, the following claim holds:*

$$\lim_{\eta \downarrow 0} \inf_{x:\boldsymbol{d}(x,\mathcal{G}^c) > \epsilon} \mathbb{P}_x \Big( X_n^\eta \in \mathcal{G} \; \forall n \leq T_{return}(\eta, \epsilon) \, and \, T_{\text{return}}(\eta, \epsilon) \leq \rho(\epsilon)/\eta \Big) = 1$$

*where the stopping time involved is defined as*

$$T_{return}(\eta, \epsilon) \triangleq \min\{n \geq 0 : \|X_n^\eta\| < 2\epsilon\}$$

*and the function $\rho(\epsilon) = c_1 + c_1 \log(2/\epsilon)$ with $c_1 \in (0, \infty)$ being the constant specified in Lemma J.4.*

*Proof.* Let $t = \rho(\epsilon) = c_1 + c_1 \log(2/\epsilon)$ and $n = \lceil t/\eta \rceil$ and arbitrarily choose some $x$ with $\boldsymbol{d}(x, \mathcal{G}^c) > \epsilon$. Also, fix some $N > 0$. Due to Lemma J.10, we are able to pick some $\delta > 0$ such that (note that $\eta n \leq 2t$ eventually as $\eta \downarrow 0$)

$$\mathbb{P}\Big( \max_{n=1,2,\cdots,\lceil t/\eta \rceil} \eta \Big\| Z_1^{\leq \delta/\eta} + \cdots + Z_n^{\leq \delta/\eta} \Big\| > \frac{\epsilon}{2\exp(2\eta nM)} \Big) = o(\eta^N)$$

First, from Lemma J.5 we know that for any $\eta$ small enough such that

$$2\eta M \exp(2tM) < \epsilon/2,$$

we must have

$$\Big\| \boldsymbol{x}^\eta(s, x) - \boldsymbol{y}_{\lfloor s \rfloor}^\eta(x) \Big\| \leq \epsilon/2 \quad \forall s \in [0, n].$$

Next, using Lemma J.8, we see that on event

$$A(n, \eta, \frac{\epsilon}{2\exp(\eta nM)}, \delta) \bigcap \{T_1^\eta(\delta) > \lceil t/\eta \rceil\},$$

we must have

$$\|\boldsymbol{y}_k^\eta(x) - X_k^\eta(x)\| < \epsilon \quad \forall k = 0, 1, \cdots, \lceil t/\eta \rceil.$$

Combining these facts with the Lemma J.4, we have shown that, for all $\eta \in (0, \frac{\epsilon}{4M\exp(2tM)})$, on event $A(n, \eta, \frac{\epsilon}{2\exp(\eta nM)}, \delta) \bigcap \{T_1^\eta(\delta) > \lceil t/\eta \rceil\}$ we must have $X_n^\eta \in \mathcal{G} \; \forall n \leq T_{\text{return}}(\eta, \epsilon)$ and $T_{\text{return}}(\eta, \epsilon) \leq \rho(\epsilon)/\eta$. In other words,

$$\lim_{\eta \downarrow 0} \sup_{x:\boldsymbol{d}(x,\mathcal{G}^c) > \epsilon} \mathbb{P}_x \Big( X_n^\eta \notin \mathcal{G} \text{ for some } n \leq T_{\text{return}}(\eta, \epsilon) \text{ OR } T_{\text{return}}(\eta, \epsilon) > \rho(\epsilon)/\eta \Big)$$

$$\leq \lim_{\eta \downarrow 0} \mathbb{P}\big(A(n, \eta, \frac{\epsilon}{2\exp(\eta nM)}, \delta)^c\big) + \lim_{\eta \downarrow 0} \mathbb{P}(T_1^\eta(\delta) \leq \lceil t/\eta \rceil).$$

However, our choice of $\delta$ at the beginning of the proof implies that $\lim_{\eta\downarrow 0}\mathbb{P}\big(A(n,\eta,\frac{\epsilon}{2\exp(\eta n M)},\delta)^c\big)=0$. Meanwhile, from Assumption I.6, we see that

$$\mathbb{P}(T_1^\eta(\delta)\leq \lceil t/\eta\rceil)=\sum_{k=1}^{\lceil t/\eta\rceil}\big(1-\sum_{j=0}^m H_j(\delta/\eta)\big)^{k-1}\cdot\big(\sum_{j=0}^m H_j(\delta/\eta)\big)$$

$$\leq (m+1)H_1(\delta/\eta)\sum_{k=1}^{\lceil t/\eta\rceil}\big(1-H_1(\delta/\eta)\big)^{k-1}$$

for all $\eta$ sufficiently small (due to regularly varying nature of $H_j$ and $1<\alpha_1<\cdots<\alpha_m$). In particular, since $H_1\in RV_{-\alpha_1}$, for any fixed $\Delta\in(0,\alpha_1-1)$, we have (for all $\eta$ sufficiently small),

$$\mathbb{P}(T_1^\eta(\delta)\leq \lceil t/\eta\rceil)\leq (m+1)\eta^{\alpha_1-\Delta}\sum_{k=1}^{\lceil t/\eta\rceil}(1-\eta^{\alpha_1+\Delta})^{k-1}=\frac{m+1}{\eta^{2\Delta}}\mathbb{P}(U(\eta)\leq \lceil t/\eta\rceil)$$

where $U(\eta)\stackrel{d}{=}Geom(\eta^{\alpha_1+\Delta})$. Lemma G.4 then tells us that, for all $\eta$ sufficiently small, $\mathbb{P}(U(\eta)\leq \lceil t/\eta\rceil)\leq\frac{2}{t}\eta^{\alpha_1-1+\Delta}$, thus implying

$$\limsup_{\eta\downarrow 0}\mathbb{P}(T_1^\eta(\delta)\leq \lceil t/\eta\rceil)\leq\limsup_{\eta\downarrow 0}\frac{2(m+1)}{t}\eta^{\alpha_1-1-\Delta}=0$$

and concluding the proof. $\square$

In the next result, we show that, once entering a $\epsilon-$small neighborhood of the local minimum, the SGD iterates will most likely stay there until the next large jump.

**Lemma J.12.** *Given $N_0>0$, the following claim holds for all $\epsilon\in(0,\bar\epsilon/3)$ and all $\delta>0$ that is sufficiently small:*

$$\sup_{x\in B(\mathbf{0},2\epsilon)}\mathbb{P}\Big(\exists n<T_1^\eta(\delta)\ s.t.\ \|X_n^\eta(x)\|>3\epsilon\Big)=o(\eta^{N_0})$$

*as $\eta\downarrow 0$.*

*Proof.* Let $t=c_1+c_1\log(1/\epsilon)$ where $c_1\in(0,\infty)$ is the constant specified in Lemma J.4, and $n=\lceil t/\eta\rceil$. Fix some $\beta>1+\alpha_1$ and $N>\beta-1+N_0$. Due to Lemma J.10, the following claim holds for all $\delta$ that is sufficiently small:

$$\mathbb{P}\Big(\max_{k=1,2,\cdots,\lceil t/\eta\rceil}\eta\big\|Z_1^{\leq\delta/\eta}+\cdots+Z_k^{\leq\delta/\eta}\big\|>\frac{\epsilon/2}{2\exp(2\eta n M)}\Big)=o(\eta^N)$$

We fix one of such $\delta$. Therefore, for event

$$\widetilde{A}\triangleq\Big\{\exists j\leq\lceil\frac{1/\eta^\beta}{\lceil t/\eta\rceil}\rceil\ s.t.$$

$$\max_{k=1,2,\cdots,\lceil t/\eta\rceil}\eta\big\|Z_{1+(j-1)\lceil t/\eta\rceil}^{\leq\delta/\eta}+Z_{2+(j-1)\lceil t/\eta\rceil}^{\leq\delta/\eta}+\cdots+Z_k^{\leq\delta/\eta}\big\|>\frac{\epsilon/2}{2\exp(2\eta n M)}\Big\},$$

the iid nature of noises $(Z_n)_{n\geq 1}$ implies that

$$\mathbb{P}(\widetilde{A})\leq(1+\frac{1}{t\eta^{\beta-1}})\mathbb{P}\Big(\max_{k=1,2,\cdots,\lceil t/\eta\rceil}\eta\big\|Z_1^{\leq\delta/\eta}+\cdots+Z_k^{\leq\delta/\eta}\big\|>\frac{\epsilon/2}{2\exp(2\eta n M)}\Big)$$

$$=o(\eta^{N-\beta+1})\ \text{as }\eta\downarrow 0.$$

Meanwhile, Note that $T_1^\eta(\delta)\leq T_1^{\eta,(1)}(\delta)\triangleq\min\{n\geq 0:\ \eta|Z_n|>\delta,\ Z_n\in E_1\}$. Therefore, $\mathbb{P}(T_1^\eta(\delta)>1/\eta^\beta)\leq\mathbb{P}(T_1^{\eta,(1)}(\delta)>1/\eta^\beta)$. Specifically, note that $T_1^\eta(\delta)\stackrel{d}{=}Geom(H_1(\delta/\eta))$ and $H_1\in RV_{-\alpha_1}$. Lemma G.3 then tells us the existence of some $c>0$ such that

$$\mathbb{P}(T_1^\eta(\delta)>1/\eta^\beta)=o\big(\exp(-c/\eta)\big)$$

as $\eta \downarrow 0$. In summary, we have established that

$$\mathbb{P}(\widetilde{A} \cup \{T_1^\eta(\delta) > 1/\eta^\beta\}) = o(\eta^{N_0}).$$

Now let us focus on the complementary event $(\widetilde{A})^c \cap \{T_1^\eta(\delta) \leq 1/\eta^\beta\}$. On this event, there must be some $j \leq \lceil \frac{1/\eta^\beta}{\lceil t/\eta \rceil} \rceil$ such that $1 + (j-1)\lceil t/\eta \rceil \leq T_1^\eta(\delta) \leq j\lceil t/\eta \rceil$.

Let us arbitrarily choose some $x \in B(\mathbf{0}, 2\epsilon)$. If $j = 1$, then due to eq. (J.3), we know that $\boldsymbol{x}^\eta(s, x) \in B(\mathbf{0}, 2\epsilon)$ for all $s \geq 0$. Also, from Lemma J.5 we know that for any $\eta$ small enough such that $2\eta M \exp(2tM) < \epsilon/2$, we must have

$$\left\| \boldsymbol{x}^\eta(s, x) - \boldsymbol{y}_{\lfloor s \rfloor}^\eta(x) \right\| \leq \epsilon/2 \quad \forall s \in [0, \lceil t/\eta \rceil].$$

Besides, on event $(\widetilde{A})^c \cup \{T_1^\eta(\delta) \leq 1/\eta^\beta\} \cap \{j = 1\}$, due to Lemma J.8, we must have

$$\|\boldsymbol{y}_k^\eta(x) - X_k^\eta(x)\| < \epsilon/2 \quad \forall k = 0, 1, \cdots, T_1^\eta(\delta)$$

with $T_1^\eta(\delta) < \lceil t/\eta \rceil$. In conclusion, in the case that $j = 1$, we must have $\|X_n^\eta(x)\| < 3\epsilon$ for all $n < T_1^\eta(\delta)$.

Now consider the case with $j \geq 2$. Similarly, due to Lemma J.5 and J.8, we now have that

$$\left\| \boldsymbol{x}^\eta(s, x) - \boldsymbol{y}_{\lfloor s \rfloor}^\eta(x) \right\| \leq \epsilon/2 \quad \forall s \in [0, \lceil t/\eta \rceil],$$
$$\|\boldsymbol{y}_k^\eta(x) - X_k^\eta(x)\| < \epsilon/2 \quad \forall k = 0, 1, \cdots, \lceil t/\eta \rceil,$$
$$\|\boldsymbol{x}^\eta(s, x)\| < 2\epsilon \quad \forall s \in [0, \lceil t/\eta \rceil].$$

In particular, due to our choice of $t$ at the beginning of the proof and Lemma J.4, one can see that $\|\boldsymbol{x}^\eta(\lceil t/\eta \rceil, x)\| < \epsilon$. Collecting all the results, we now know that

$$\|X_k^\eta(x)\| < 3\epsilon \ \forall k \leq \lceil t/\eta \rceil, \ \left\| X_{\lceil t/\eta \rceil}^\eta(x) \right\| < 2\epsilon.$$

Now it suffices to apply the repeated apply the same arguments above. For instance, if $j = 2$, then by letting $x_1 = X_{\lceil t/\eta \rceil}^\eta(x)$ and bounding the gap between trajectory of $X_{\lfloor s \rfloor + \lceil t/\eta \rceil}^\eta(x)$ and $\boldsymbol{x}^\eta(s, x_1)$ using Lemma J.5 and J.8, one can show that $\|X_n^\eta(x)\| < 3\epsilon$ for all $n < T_1^\eta(\delta)$ (since in the case of $j = 2$, we have $T_1^\eta(\delta) < 2\lceil t/\eta \rceil$). Otherwise, we have $j \geq 3$ and the same arguments above can be applied to show that

$$\|X_k^\eta(x)\| < 3\epsilon \ \forall k \leq 2\lceil t/\eta \rceil, \ \left\| X_{\lceil t/\eta \rceil}^\eta(x) \right\| < 2\epsilon, , \ \left\| X_{2\lceil t/\eta \rceil}^\eta(x) \right\| < 2\epsilon.$$

In summary, by proceeding inductively, we establish that for all $\eta \in (0, \frac{\epsilon}{4M \exp(2tM)})$,

$$\sup_{x \in B(\mathbf{0}, 2\epsilon)} \mathbb{P}\Big( \exists n < T_1^\eta(\delta) \ s.t. \ \|X_n^\eta(x)\| > 3\epsilon \Big) \leq \mathbb{P}(\widetilde{A} \cup \{T_1^\eta(\delta) > 1/\eta^\beta\})$$

and this concludes the proof. $\qquad\square$

We introduce a few concepts that will be crucial to the analysis below. Recall the definition of ODE with jumps $\widetilde{\boldsymbol{x}}$ (note that we will drop the notational dependency on learning rate $\eta$ when we choose $\eta = 1$). Recall the definitions of $\boldsymbol{i}^*, k^*$ (see Assumption I.7 and I.8 and the remarks underneath). We define the following mapping $h$ from $\boldsymbol{r} = (r_1, \cdots, r_{k^*}), \boldsymbol{\theta} = (\theta_1, \cdots, \theta_{k^*}) \in (\mathbb{S}^{d-1})^{k^*}, \boldsymbol{t} = (t_1, \cdots, t_{k^*}) \in \{0\} \times \mathbb{R}_+^{k^*}$ such that

$$h(\boldsymbol{r}, \boldsymbol{\theta}, \boldsymbol{t}) = \widetilde{\boldsymbol{x}}(\sum_{i=1}^{k^*} t_i, \mathbf{0}; \ \boldsymbol{t}, \boldsymbol{w}) \tag{J.38}$$

where $\boldsymbol{w} = (w_i)_{i=1}^{k^*}$ with $w_i = r_i \theta_i$. From the continuity of the ODE flow, we see that $h$ is a continuous mapping. Also, we introduce the concept of *type* for configuration $\boldsymbol{i}^*$. Specifically, define

$$\boldsymbol{j}(i_1, \cdots, i_m) \triangleq \left\{ (j_1, \cdots, j_{k^*}) \in \{0, 1, 2, \cdots, m\}^{k^*} : \#\{n : j_n = k\} = i_k \ \forall k \in [m] \right\} \tag{J.39}$$

and for any $\boldsymbol{j} \in \boldsymbol{j}(\boldsymbol{i}^*)$, we say that $\boldsymbol{w} = (w_1, \cdots, w_{k^*}) \in \mathcal{A}(\boldsymbol{i}^*)$ has type $\boldsymbol{j}$ if
$$w_i \in E_{j_i} \; \forall i \in [k^*].$$
In other words, based on the direction of each jump in $\boldsymbol{w} \in \mathcal{A}(\boldsymbol{i}^*)$ we group them into different types. Note that $|\boldsymbol{j}(\boldsymbol{i}^*)| < \infty$. Now for any type $\boldsymbol{j} \in \boldsymbol{j}(\boldsymbol{i}^*)$, define a (Borel) measure on $\mathbb{R}_+^{k^*} \times (\mathbb{S}^{d-1})^{k^*} \times \mathbb{R}_+^{k^*-1}$ as

$$\mu_{\boldsymbol{j}} \triangleq \Big(\prod_{i=1}^{k^*} \nu_{\alpha_{j_i}}\Big) \times \Big(\prod_{i=1}^{k^*} S_{\boldsymbol{j}_i}\Big) \times \mathbf{Leb}^{k^*-1} \tag{J.40}$$

where, for any $\alpha > 0$, the measure $\nu_\alpha$ is the Borel measure on $(0, \infty)$ with $\nu_\alpha(x, \infty) = 1/x^\alpha$. We use the concepts above to characterize behavior of large noises in $(Z_n)_{n \geq 1}$.

**Definition J.1.** *For any $\eta, \delta > 0, \boldsymbol{j} \in \boldsymbol{j}(\boldsymbol{i}^*)$ and any Borel set $A \subseteq \mathbb{R}^d$, we say noise $Z_n$ is of* **type-**$(A, \delta, \eta, \boldsymbol{j})$ *iff*

- *$\eta \|Z_n\| > \delta$;*

- *For $\widetilde{t}_1 = n$ and $\widetilde{t}_i \triangleq \min\{k > \widetilde{t}_{i-1} : \eta \|Z_k\| > \delta\}$ for all $i = 2, 3, \cdots, k^*$, it holds that $\widetilde{t}_i - \widetilde{t}_{i-1} < 2\bar{t}/\eta$ for all $i = 2, 3, \cdots, k^*$;*

- *For $t_i \triangleq \widetilde{t}_i - \widetilde{t}_{i-1}$ (with $t_1 = 0$), $w_i = \eta Z_{\widetilde{t}_i}$, and*
$$\boldsymbol{r} \triangleq (\|w_1\|, \|w_2\|, \cdots, \|w_{k^*}\|),$$
$$\boldsymbol{\theta} = (w_1/\|w_1\|, \cdots, w_{k^*}/\|w_{k^*}\|),$$
$$\boldsymbol{t} = (t_1, \cdots, t_{k^*}),$$
  *it holds that $h(\boldsymbol{r}, \boldsymbol{\theta}, \boldsymbol{t}) \in A$.*

*More generally, we say noise $Z_n$ is of* **type-**$(A, \delta, \eta)$ *iff there exists $\boldsymbol{j} \in \boldsymbol{j}(\boldsymbol{i}^*)$ such that $Z_n$ is of* **type-**$(A, \delta, \eta, \boldsymbol{j})$.

Due to the iid nature of $(Z_j)_{j \geq 1}$, let us consider an iid sequence $(V_j)_{j \geq 0}$ where the sequence has the same law of $Z_1$. Note that for any fixed $n \geq 1$, the probability that $Z_n$ is of type-$(A, \delta, \eta, \boldsymbol{j})$ is equal to the probability that $V_0$ is of type-$(A, \delta, \eta, \boldsymbol{j})$. More specifically, let $H(x) = \sum_{j=0}^m H_j(x) = \mathbb{P}(\|Z_1\| > x)$. Due to Assumption I.6, $H \in RV_{-\alpha_1}$. Besides, $\mathbb{P}(\eta \|V_0\| > \delta) = H(\delta/\eta)$. Now we focus on conditional probability admitting the following form:

$$p(A, \delta, \eta, \boldsymbol{j}) = \mathbb{P}\Big(V_0 \text{ is of type-}(A, \delta, \eta, \boldsymbol{j}) \,\Big|\, \eta \|V_0\| > \delta\Big).$$

**Lemma J.13.** *For any $\delta > 0$, any $\boldsymbol{j} \in \boldsymbol{j}(\boldsymbol{i}^*)$, and any Borel set $A \subseteq \mathbb{R}^d$ such that*
$$h(r_1, \cdots, r_{k^*}, \theta_1, \cdots, \theta_{k^*}, t_2, \cdots, t_{k^*}) \in A \implies t_i < 2\bar{t} \; \forall i = 2, \cdots, k^* \text{ and } r_i > \delta \; \forall i \in [k^*], \tag{J.41}$$

*it holds that*

$$\mu_{\boldsymbol{j}}\big(h^{-1}(A^\circ)\big) \leq \liminf_\eta \frac{p(A, \delta, \eta, \boldsymbol{j})}{\delta^{\alpha_1} \cdot \frac{\prod_{k=1}^m \big(H_k(1/\eta)\big)^{i_k^*}}{H(1/\eta)\eta^{k^*-1}}}$$

$$\leq \limsup_\eta \frac{p(A, \delta, \eta, \boldsymbol{j})}{\delta^{\alpha_1} \cdot \frac{\prod_{k=1}^m \big(H_k(1/\eta)\big)^{i_k^*}}{H(1/\eta)\eta^{k^*-1}}} \leq \mu_{\boldsymbol{j}}\big(h^{-1}(\bar{A})\big).$$

*Proof.* Let us start by fixing some notations. Let $T_1 = 0$, and define stopping times $T_j = \min\{n > T_{j-1} : \eta \|V_n\| > \delta\}$ and inter-arrival times $T_j' = T_j - T_{j-1}$ for any $j \geq 2$ (with $T_1' = 0$), and large jump $W_j = V_{T_j}$ for any $j \geq 0$. Note that: first, the pair $(T_i', W_i)$ is independent of $(T_j', W_j)$ for any $i \neq j$; besides, $W_j$ and $T_j'$ are independent for all $j \geq 1$.

Define the following sequence (of random elements) $\boldsymbol{w} = (w_1, \cdots, w_{l^*})$ and $\boldsymbol{t} = (t_1, \cdots, t_{l^*})$ by
$$w_i = \eta W_i, \quad t_i = \eta T_i'.$$
and $\boldsymbol{r} = (\mathbf{T}_r(w_i))_{i=1}^{k^*}, \boldsymbol{\theta} = (\mathbf{T}_\theta(w_i))_{i=1}^{k^*}$ where operators $\mathbf{T}_r(x) = \|x\|, \mathbf{T}_\theta(x) = x/\|x\|$ constitute the polar coordinate transform. By definition of type-$(A, \delta, \eta, \boldsymbol{j})$, we know that

- $T_i' \leq 2\bar{t}/\eta$ for any $i = 2, \cdots, k^*$;

- $\eta \|W_i\| > \delta$ for any $i = 1, 2, \cdots, k^*$;

- $w_i \in E_{\boldsymbol{j}_i}$ (i.e. $\theta_i \in F_i$) for all $i \in [k^*]$;

- $h(\boldsymbol{r}, \boldsymbol{\theta}, \boldsymbol{t}) \in A$.

Therefore, (let $R_i = \mathbf{T}_r(W_i), \Theta_i = \mathbf{T}_\theta(W_i)$)

$$
\begin{aligned}
&p(A, \delta, \eta, \boldsymbol{j}) \\
&= \Big( \mathbb{P}(T_2' \leq 2\bar{t}/\eta) \Big)^{k^*-1} \cdot \int \mathbb{1}\Big\{ (\boldsymbol{r}, \boldsymbol{\theta}, \boldsymbol{t}) \in h^{-1}(A) \Big\} \\
&\qquad\qquad \cdot \Big( \prod_{i=1}^{k^*} \mathbb{P}(W_i \in E_i) \cdot \mathbb{P}(\eta R_i = dr_i, \ \Theta_i = d\theta_i) \Big) \cdot \Big( \prod_{i=2}^{k^*} \mathbb{P}(\eta T_i' = dt_i \mid \eta T_i' \leq 2\bar{t}) \Big) \\
&= \Big( \mathbb{P}(T_2' \leq 2\bar{t}/\eta) \Big)^{k^*-1} \cdot \Big[ \prod_{i=1}^{k^*} \mathbb{P}(W_i \in E_i) \Big] \cdot \mathbb{Q}_{\eta,\delta,\boldsymbol{j}}\big( h^{-1}(A) \big)
\end{aligned}
\tag{J.42}
$$

where $\mathbb{Q}_{\eta,\delta,\boldsymbol{j}}$ is the probability measure on $\mathbb{R}_+^{k^*} \times (\mathbb{S}^{d-1})^{k^*} \times \mathbb{R}_+^{k^*-1}$ induced by a sequence of random elements $(\eta R_1, \cdots, \eta R_{k^*}, \Theta_1, \cdots, \Theta_{k^*}, \eta T_2^\uparrow(\eta), \cdots, \eta T_{k^*}^\uparrow(\eta))$ such that

- For any $i = 1, \cdots, k^*$, we have $R_i = \mathbf{T}_r(W_i^\uparrow(\eta)), \Theta_i = \mathbf{T}_\theta(W_i^\uparrow(\eta))$ where the distribution of $W_i^\uparrow(\eta)$ follows the law $\mathbb{P}\Big( V_0 \in \cdot \ \Big| \ \eta \|V_0\| > \delta, \ V_0 \in E_{\boldsymbol{j}_i} \Big)$; Besides, $(W_i^\uparrow(\eta))_{i=1}^{k^*}$ is an independent sequence;

- For any $i = 2, \cdots, l^*$, the distribution of $T_i^\uparrow(\eta)$ follows from $\mathbb{P}\Big( \eta T_1^\delta(\eta) \in \cdot \ \Big| \ \eta T_1^\delta(\eta) \leq 2\bar{t} \Big)$ (for the definition of stopping times $T_k^\delta(\eta)$, see eq. (J.26)); Besides, $(T_i^\uparrow(\eta))_{i=1}^{k^*}$ is an independent sequence, and it is independent of $(W_i^\uparrow(\eta))_{i=1}^{k^*}$;

- $\mathbb{Q}_{\eta,\delta,\boldsymbol{j}}(\cdot) = \mathbb{P}\Big( (\eta R_1, \cdots, \eta R_{k^*}, \Theta_1, \cdots, \Theta_{k^*}, \eta T_2^\uparrow(\eta), \cdots, \eta T_{k^*}^\uparrow(\eta)) \in \cdot \Big)$.

As for weak convergence of $(\eta R_1, \cdots, \eta R_{k^*}, \Theta_1, \cdots, \Theta_{k^*}, \eta T_2^\uparrow(\eta), \cdots, \eta T_{k^*}^\uparrow(\eta))$, observe that

- For any $i \in [k^*]$, from Assumption I.6 one can see the regularly varying nature of distribution of $V_0$ on the cone $E_{\boldsymbol{j}_i}$ (hence for $W_i^\uparrow(\eta)$ as well), thus yielding that $(\eta R_i, \Theta_i) \Rightarrow (R_i^*, \Theta_i^*)$ as $\eta \downarrow 0$ where $R_i^*$ and $\Theta_i^*$ are independent, the law of $\Theta_i^*$ is $S_{\boldsymbol{j}_i}$, and the law of $R_i^*$ is the Pareto distribution with

$$
\mathbb{P}(R_i^* > x) = \frac{\delta^{\alpha_{j_i}}}{(x \vee \delta)^{\alpha_{j_i}}};
$$

- For any $x \in [0, 2\bar{t}]$, since $\lim_{\eta\downarrow 0} \lfloor x/\eta \rfloor H(\delta/\eta) = 0$, it is easy to show that

$$
\lim_{\eta\downarrow 0} \frac{1 - (1 - H(\delta/\eta))^{\lfloor x/\eta \rfloor}}{\lfloor x/\eta \rfloor H(\delta/\eta)} = 1;
$$

therefore, we have (for any $x \in (0, 2\bar{t}]$)

$$
\mathbb{P}(\eta T_1^\delta(\eta) \leq x \mid \eta T_1^\delta(\eta) \leq 2\bar{t}) = \frac{1 - (1 - H(\delta/\eta))^{\lfloor x/\eta \rfloor}}{1 - (1 - H(\delta/\eta))^{\lfloor 2\bar{t}/\eta \rfloor}} \to \frac{x}{2\bar{t}}
$$

as $\eta \downarrow 0$, which implies that $T_i^\uparrow(\eta)$ converges weakly to a uniform RV on $[0, 2\bar{t}]$.

Together with the assumption on set $A$ in eq. (J.41), we can now see that, if we denote the weak limit of measure $\mathbb{Q}_{\eta,\delta,\boldsymbol{j}}$ as $\mu_{\delta,\boldsymbol{j}}$, then (the measure $\mu_{\boldsymbol{j}}$ is defined in eq. (J.40))

$$\mu_{\delta,\boldsymbol{j}}\big(h^{-1}(A)\big) = \frac{\prod_{i=1}^{k^*} \delta^{\alpha_{j_i}}}{(2\bar{t})^{k^*-1}} \mu_{\boldsymbol{j}}\big(h^{-1}(A)\big).$$

From the continuity of mapping $h$, one can see that $h^{-1}(\bar{A})$ is a closed set and $h^{-1}(A^\circ)$ is an open set. Using Portmanteau theorem, we now have

$$\frac{\prod_{i=1}^{k^*} \delta^{\alpha_{j_i}}}{(2\bar{t})^{k^*-1}} \mu_{\boldsymbol{j}}\big(h^{-1}(A^\circ)\big) \leq \liminf_{\eta} \mathbb{Q}_{\eta,\delta,\boldsymbol{j}}\big(h^{-1}(A)\big) \leq \limsup_{\eta} \mathbb{Q}_{\eta,\delta,\boldsymbol{j}}\big(h^{-1}(A)\big) \tag{J.43}$$

$$\leq \frac{\prod_{i=1}^{k^*} \delta^{\alpha_{j_i}}}{(2\bar{t})^{k^*-1}} \mu_{\boldsymbol{j}}\big(h^{-1}(\bar{A})\big). \tag{J.44}$$

Moving on, we analyze the limit of the other terms in eq. (J.42). For any $i \in [k^*]$, note that

$$\mathbb{P}(W_i \in E_i) = \frac{H_{\boldsymbol{j}_i}(\delta/\eta)}{H(\delta/\eta)}.$$

Now due to the regularly varying nature of $H_j$ and $H$,

$$\lim_{\eta} \frac{\prod_{i=1}^{k^*} \mathbb{P}(W_i \in E_i)}{\prod_{i=1}^{k^*} H_{\boldsymbol{j}_i}(1/\eta)/H(1/\eta)} = \frac{\delta^{k^* \alpha_1}}{\prod_{i=1}^{k^*} \delta^{\alpha_{j_i}}}. \tag{J.45}$$

On the other hand, from Lemma G.4 and the regularly varying nature of function $H$, for any fixed $\kappa > 1$, we have

$$\limsup_{\eta\downarrow 0} \Big(\frac{\delta^{\alpha_1}}{2\bar{t}} \cdot \frac{\mathbb{P}(T_2' \leq 2\bar{t}/\eta)}{H(1/\eta)/\eta}\Big)^{k^*-1} \leq \kappa^{k^*-1} \limsup_{\eta\downarrow 0} \Big(\frac{\delta^{\alpha_1}}{2\bar{t}} \cdot \frac{2\bar{t}H(\delta/\eta)/\eta}{H(1/\eta)/\eta}\Big)^{k^*-1} = \kappa^{k^*-1},$$

$$\liminf_{\eta\downarrow 0} \Big(\frac{\delta^{\alpha_1}}{2\bar{t}} \cdot \frac{\mathbb{P}(T_2' \leq 2\bar{t}/\eta)}{H(1/\eta)/\eta}\Big)^{k^*-1} \geq (1/\kappa)^{k^*-1} \liminf_{\eta\downarrow 0} \Big(\frac{\delta^{\alpha_1}}{2\bar{t}} \cdot \frac{2\bar{t}H(\delta/\eta)/\eta}{H(1/\eta)/\eta}\Big)^{k^*-1}$$

$$= (1/\kappa)^{k^*-1}.$$

Due to arbitrariness of $\kappa > 1$, we yield that

$$\lim_{\eta} \frac{\big(\mathbb{P}(T_1' \leq 2\bar{t}/\eta)\big)^{k^*-1}}{\Big(\frac{H(1/\eta)}{\eta} \cdot \frac{2\bar{t}}{\delta^{\alpha_1}}\Big)^{k^*-1}} = 1. \tag{J.46}$$

Collecting all the limits in eq. (J.44)-eq. (J.46) and plugging them into eq. (J.42), we now have established that

$$\mu_{\boldsymbol{j}}\big(h^{-1}(A^\circ)\big) \leq \liminf_{\eta} \frac{p(A,\delta,\eta,\boldsymbol{j})}{\delta^{\alpha_1} \cdot \frac{\prod_{i=1}^{k^*} H_{\boldsymbol{j}_i}(1/\eta)}{H(1/\eta)\eta^{k^*-1}}} \leq \limsup_{\eta} \frac{p(A,\delta,\eta,\boldsymbol{j})}{\delta^{\alpha_1} \cdot \frac{\prod_{i=1}^{k^*} H_{\boldsymbol{j}_i}(1/\eta)}{H(1/\eta)\eta^{k^*-1}}} \leq \mu_{\boldsymbol{j}}\big(h^{-1}(\bar{A})\big).$$

To conclude the proof, recall that for any type $\boldsymbol{j} \in \boldsymbol{j}(\boldsymbol{i}^*)$, we have $\#\{k = 1, 2, \cdots, k^* : \boldsymbol{j}_k = j\} = i_j^*$ (i.e. for any $j \in [m]$, the number of elements in $(\boldsymbol{j}_1, \cdots, \boldsymbol{j}_{k^*})$ that are equal to $j$ is exactly $i_j^*$), thus $\prod_{i=1}^{k^*} H_{\boldsymbol{j}_i}(1/\eta) = \prod_{k=1}^{m} \big(H_k(1/\eta)\big)^{i_k^*}$. $\qquad\square$

More generally, for

$$p(A,\delta,\eta) = \mathbb{P}\Big(V_0 \text{ is of type-}(A,\delta,\eta) \ \Big| \ \eta\|V_0\| > \delta\Big), \tag{J.47}$$

we know that $p(A,\delta,\eta) = \sum_{\boldsymbol{j}\in\boldsymbol{j}(\boldsymbol{i}^*)} p(A,\delta,\eta,\boldsymbol{j})$. Also, define measure $\mu$ as

$$\mu = \sum_{\boldsymbol{j}\in\boldsymbol{j}(\boldsymbol{i}^*)} \mu_{\boldsymbol{j}}. \tag{J.48}$$

The next result follows immediately from Lemma J.13 and the fact that $|\boldsymbol{j}(\boldsymbol{i}^*)| < \infty$. Recall that $J_{\mathcal{G}}$ is defined in eq. (I.9).

**Corollary J.14.** *There exists a function* $\widetilde{\lambda}(\eta) \in RV_{1+J_{\mathcal{G}}-\alpha_1}(\eta)$ *such that, given any* $\delta > 0$ *and any Borel set* $A \subseteq \mathbb{R}^d$ *with*

$$h(r_1, \cdots, r_{k^*}, \theta_1, \cdots, \theta_{k^*}, t_2, \cdots, t_{k^*}) \in A \implies t_i < 2\bar{t} \ \forall i = 2, \cdots, k^* \text{ and } r_i > \delta \ \forall i \in [k^*],$$
(J.49)

*it holds that*

$$\mu\big(h^{-1}(A^\circ)\big) \leq \liminf_\eta \frac{p(A, \delta, \eta)}{\delta^{\alpha_1} \cdot \widetilde{\lambda}(\eta)} \leq \limsup_\eta \frac{p(A, \delta, \eta)}{\delta^{\alpha_1} \cdot \widetilde{\lambda}(\eta)} \leq \mu\big(h^{-1}(\bar{A})\big).$$

*In particular, the regularly varying function* $\widetilde{\lambda}(\cdot)$ *admits the form*

$$\widetilde{\lambda}(\eta) = \frac{\prod_{j=1}^m \big(H_j(1/\eta)\big)^{i_j^*}}{H(1/\eta)\eta^{k^*-1}}$$
(J.50)

Consider the following stopping times

$$\sigma(\eta) = \min\{n \geq 0 : X_n^\eta \notin \mathcal{G}\};$$
$$R(\epsilon, \delta, \eta) = \min\{n \geq T_1^\eta(\delta) : \|X_n^\eta\| \leq 3\epsilon\}.$$

Here $\sigma$ indicate the first time that the iterates exit domain $\mathcal{G}$, while $R$ denotes the time the SGD iterates return to a small neighborhood of the attractor $\mathbf{0}$ after the first large jump. In the next few results, we study the probability of the different scenarios regarding the first exit time $\sigma(\eta)$ and first return time $R(\epsilon, \delta, \eta)$. To this end, let

$$K^\eta(\delta) \triangleq \max\{k \geq 0 : T_k^\eta(\delta) \leq \sigma(\eta) \wedge R(\epsilon, \delta, \eta)\}$$

be the count of large jumps before the first exit or first return. Furthermore, we introduce the following concept as the *accumulated cost* of large jumps before the first exit or return. Let $\mathcal{J}_0^{\eta,\delta} = 0$ and

$$\mathcal{J}_k^{\eta,\delta} \triangleq \mathcal{J}_{k-1}^{\eta,\delta} + J(W_k^\eta(\delta)) \quad \forall k \leq K^\eta(\delta)$$

where the cost function $J(\cdot)$ is defined in eq. (I.6). In other words, $\mathcal{J}_k^{\eta,\delta}$ indicates the total cost of the first $k$ large jumps, if there are at least $k$ large jumps, before the first exit or first return. Similarly, we can also define a step-wise accumulated cost as

$$\mathcal{J}_{\eta,\delta}^\downarrow(n) = \max\{\mathcal{J}_k^{\eta,\delta} : T_k^\eta(\delta) \leq n\} \ \forall n \leq \sigma(\eta) \wedge R(\epsilon, \delta, \eta).$$

In other words, $\mathcal{J}_{\eta,\delta}^\downarrow(n)$ evaluates the total cost of large jumps up until step $n$. As a preparation for our analyses below, we first discuss the following technical tools. For any $\epsilon > 0$, let $\hat{t}(\epsilon) = c_1 + c_1 \log(1/\epsilon)$ where $c_1$ is the constant in Lemma J.4. Besides, define

$$\widetilde{\epsilon}(\epsilon) \triangleq \frac{\epsilon}{4\exp(2\hat{t}M)} \wedge \frac{\epsilon}{\bar{\rho}(\hat{t}(\epsilon))} \wedge \frac{\epsilon}{\widetilde{\rho}(\hat{t}(\epsilon))}$$
(J.51)

where functions $\bar{\rho}(\cdot), \widetilde{\rho}(\cdot)$ are defined in Corollary J.6 and J.9 respectively. Furthermore, define an event $A^\times = A_1^\times(\epsilon, \delta, \eta) \cup A_2^\times(\epsilon, \delta, \eta)$ where

$$A_1^\times(\epsilon, \delta, \eta) \triangleq \Big\{\exists n < T_1^\eta(\delta) \text{ s.t. } \|X_n^\eta(x)\| > 3\epsilon\Big\},$$
(J.52)

$$A_2^\times(\epsilon, \delta, \eta) \triangleq \Big\{\exists j = 2, \cdots, l^* \text{ s.t. }$$

$$\max_{k=1,2,\cdots,(T_j^\eta(\delta)-T_{j-1}^\eta(\delta)-1)\wedge(2\hat{t}(\epsilon)/\eta)} \eta\Big\|Z_{T_{j-1}^\eta(\delta)+1} + \cdots + Z_{T_{j-1}^\eta(\delta)+j}\Big\| > \widetilde{\epsilon}(\epsilon)\Big\}$$
(J.53)

where the positive integer $l^*$ is defined in eq. (J.1). By definition of $l^*$, we must have $K^\eta(\delta) < l^*$. Lastly, define events (for any $K \in [l^*]$)

$$B_1^\times(K) \triangleq (A^\times)^c \cap \{\mathcal{J}_K^{\eta,\delta} < J_{\mathcal{G}}, \ K \leq K^\eta(\delta)\}$$

$$\cap \{\forall j \in [K], \ [T_{j+1}^\eta(\delta) \wedge \sigma(\eta) \wedge R(\epsilon, \delta, \eta)] - T_j^\eta(\delta) \leq \frac{2\hat{t}(\epsilon)}{\eta}\},$$

$$B_2^\times(K) \triangleq (A^\times)^c \cap \{\mathcal{J}_K^{\eta,\delta} \leq J_{\mathcal{G}}, \ K \leq K^\eta(\delta)\}$$

$$\cap \{\exists j \in [K] \text{ s.t. } [T_{j+1}^\eta(\delta) \wedge \sigma(\eta) \wedge R(\epsilon, \delta, \eta)] - T_j^\eta(\delta) > \frac{2\hat{t}(\epsilon)}{\eta}\}.$$

Recall that the minimum cost for exit $J_\mathcal{G}$ is defined in eq. (I.9). Note that, in the definition of $B_1^\times(K)$ and $B_2^\times(K)$ above, $T_{j+1}^\eta(\delta) \wedge \sigma(\eta) \wedge R(\epsilon, \delta, \eta) < T_{j+1}^\eta(\delta)$ only if $j = K = K^\eta(\delta)$.

**Lemma J.15.** *For any* $K \in [l^*]$*, any* $\epsilon \in (0, \bar{\epsilon}/3)$ *and any* $\eta > 0$ *sufficiently small, the following claim holds on event* $B_1^\times(K)$*:*

$$\sup_{s \in [0, T_{K+1}^\eta(\delta) \wedge \sigma(\eta) \wedge R(\epsilon, \delta, \eta) - T_1^\eta(\delta)]} \left\| X_{\lfloor s \rfloor + T_1^\eta(\delta)}^\eta(x) - \widetilde{\boldsymbol{x}}^\eta(s, X^{(1)}(x); \boldsymbol{T}, \boldsymbol{W}) \right\| < 2\epsilon$$

*where* $\boldsymbol{T} = \left(0, T_2^\eta(\delta) - T_1^\eta(\delta), T_3^\eta(\delta) - T_2^\eta(\delta), \cdots, T_{K+1}^\eta(\delta) - T_K^\eta(\delta)\right)$, $\boldsymbol{W} = \left(W_1^\eta(\delta), W_2^\eta(\delta), \cdots, W_{K+1}^\eta(\delta)\right)$, *and* $X^{(1)}(x) = X_{T_1^\eta(\delta)-1}^\eta(x)$.

*Proof.* We focus on the distances between the following three objects: $X_n^\eta(x)$,

$$\widetilde{Y}_n^\eta(x) = \begin{cases} X_n^\eta(x) & \text{if } n < T_1^\eta(\delta); \\ \widetilde{Y}_{n-1}^\eta(x) + \varphi_b\left(-\eta\nabla f\left(\widetilde{Y}_{n-1}^\eta(x)\right) + \sum_{j \geq 1} \mathbb{1}\{n = T_j^\eta(\delta)\}\eta Z_n\right) & \text{otherwise}; \end{cases}$$

and

$$\boldsymbol{z}(s) = \widetilde{\boldsymbol{x}}^\eta\left(s, X^{(1)}(x); \boldsymbol{T}, \boldsymbol{W}\right) \, \forall s \geq 0.$$

First of all, for $X^{(1)} = X_{T_1^\eta(\delta)-1}^\eta(x) = \widetilde{Y}_{T_1^\eta(\delta)-1}^\eta(x)$, by definition of event $(A^\times)^c$ we have $\left\| X^{(1)} \right\| < 3\epsilon < \bar{\epsilon}$. Then due to eq. (J.20) and the definition of event $B_1^\times(K)$ (in particular, the fact that $\mathcal{J}_K^{\eta, \delta} < J_\mathcal{G}$), we know that

$$\boldsymbol{d}(\boldsymbol{z}(s), \mathcal{G}^c) > 100l^*\bar{\epsilon} \, \forall s \in \left[0, T_{K+1}^\eta(\delta) \wedge \sigma(\eta) \wedge R(\epsilon, \delta, \eta) - T_1^\eta(\delta)\right).$$

In light of the definition of $B_1^\times(K)$ (i.e. the upper bound on $T_{j+1}^\eta(\delta) - T_j^\eta(\delta)$), by applying Corollary J.6, we can show that

$$\sup_{s \in [0, T_{K+1}^\eta(\delta) \wedge \sigma(\eta) \wedge R(\epsilon, \delta, \eta) - T_1^\eta(\delta)]} \left\| \boldsymbol{z}(s) - \widetilde{Y}_{\lfloor s \rfloor + T_1^\eta(\delta)}^\eta(x) \right\| < \epsilon.$$

In particular, by applying Corollary J.6 inductively for all $n \leq T_{K+1}^\eta(\delta) \wedge \sigma(\eta) \wedge R(\epsilon, \delta, \eta) - T_1^\eta(\delta)$, we know that the line segment between $\boldsymbol{z}(s)$ and $\widetilde{Y}_{\lfloor s \rfloor + T_1^\eta(\delta)}^\eta(x)$ is in $\mathcal{G}$ for all $s < T_{K+1}^\eta(\delta) \wedge \sigma(\eta) \wedge R(\epsilon, \delta, \eta) - T_1^\eta(\delta)$ so Corollary J.6 can be further applied to $n+1$ and establish the inequality above. Similarly, due to our choice of $\hat{\epsilon}(\epsilon)$ in eq. (J.51) and the upper bound on $T_{j+1}^\eta(\delta) - T_j^\eta(\delta)$ in definition of event $B_1^\times(K)$, by applying Corollary J.9 inductively, we know that for all $\eta$ sufficiently small,

$$\sup_{j=0,1,2,\cdots, T_{K+1}^\eta(\delta) \wedge \sigma(\eta) \wedge R(\epsilon, \delta, \eta) - T_1^\eta(\delta)} \left\| \widetilde{Y}_{\lfloor s \rfloor + T_1^\eta(\delta)}^\eta(x) - X_{\lfloor s \rfloor + T_1^\eta(\delta)}^\eta(x) \right\| < \epsilon$$

holds on event $B_1^\times(K)$ and this concludes the proof. $\square$

**Lemma J.16.** *For any* $K \in [l^*]$*, any* $\epsilon > 0$ *and any* $\eta > 0$ *sufficiently small,*

$$B_2^\times(K) = \emptyset.$$

*Proof.* The definition of event $B_2^\times(K)$ ensures that we can define some $j^*$ as the smallest integer in $[K^\eta(\delta)]$ such that for $T^* \triangleq T_{j^*}^\eta(\delta) + \lceil 2\hat{t}(\epsilon)/\eta \rceil$, we have

$$T_{j^*}^\eta(\delta) < T^* < T_{j^*+1}^\eta(\delta) \wedge \sigma(\eta) \wedge R(\epsilon, \delta, \eta). \tag{J.54}$$

Analogous to the proof of the previous lemma, we focus on the pair-wise distances between the following three objects: $X_n^\eta(x)$,

$$\widetilde{Y}_n^\eta(x) = \begin{cases} X_n^\eta(x) & \text{if } n < T_1^\eta(\delta); \\ \widetilde{Y}_{n-1}^\eta(x) + \varphi_b\left(-\eta\nabla f\left(\widetilde{Y}_{n-1}^\eta(x)\right) + \sum_{j \geq 1} \mathbb{1}\{n = T_j^\eta(\delta)\}\eta Z_n\right) & \text{otherwise}; \end{cases}$$

and

$$\boldsymbol{z}(s) = \widetilde{\boldsymbol{x}}^\eta\big(s, X^{(1)}; \boldsymbol{T}, \boldsymbol{W}\big) \ \forall s \geq 0.$$

with $\boldsymbol{T} = \big(0, T_2^\eta(\delta) - T_1^\eta(\delta), T_3^\eta(\delta) - T_2^\eta(\delta), \cdots, T_{j^*+1}^\eta(\delta) - T_{j^*}^\eta(\delta)\big)$, $\boldsymbol{W} = \big(W_1^\eta(\delta), W_2^\eta(\delta), \cdots, W_{j^*+1}^\eta(\delta)\big)$, and $X^{(1)} = X_{T_1^\eta(\delta)}^\eta$. Again, using Corollary J.6 and J.9, one can see that for all $\eta > 0$ sufficiently small, we must have

$$\sup_{s \in [0, T^* - T_1^\eta(\delta)]} \left\| X_{\lfloor s \rfloor + T_1^\eta(\delta)}^\eta(x) - \boldsymbol{z}(s) \right\| < 2\epsilon$$

on event $B_2^\times(K)$. However, eq. (J.20) and $\left\| X^{(1)} \right\| < 3\epsilon < \bar{\epsilon}$ implies that

$$\sup_{s \in [0, T_{j^*}^\eta(\delta) - T_1^\eta(\delta)]} \boldsymbol{d}\big(\boldsymbol{z}(s), \, \mathcal{G}^c\big) > 100 l^* \bar{\epsilon}.$$

In the meantime, Lemma J.4 and our choice of $\widetilde{\epsilon}(\epsilon)$ in eq. (J.51) implies that

$$\|\boldsymbol{z}(T^* - T_1^\eta(\delta))\| < \epsilon \text{ and } \|X_{T^*}^\eta(x)\| < 3\epsilon,$$

thus dictating that $R(\epsilon, \delta, \eta) < T^*$ and contradicting eq. (J.54). This concludes the proof. $\qquad\square$

Now we are ready to apply the tools above and analyze some atypical scenarios regrading the first exit and first return time. In the next result we show that, when starting from the local minimum, it is very unlikely to escape if the total cost of large jumps is less than $J_\mathcal{G}$ .

**Lemma J.17.** *For any fixed $\epsilon \in (0, \bar{\epsilon}/3), N > 0$, the following claim holds for any sufficiently small $\delta > 0$:*

$$\sup_{x: \|x\| \leq 2\epsilon} \mathbb{P}_x \Big(\sigma(\eta) < R(\epsilon, \delta, \eta), \, \mathcal{J}_{\eta,\delta}^\downarrow(\sigma(\eta)) < J_\mathcal{G}\Big) = o(\eta^N) \ \text{as } \eta \downarrow 0.$$

*Proof.* Since the constant $\epsilon > 0$ is fixed, one can see that $\hat{t} = c_1 + c_1 \log(1/\epsilon)$ is also fixed where $c_1$ is the constant in Lemma J.4. Besides, we can fix some $\widetilde{\epsilon} > 0$ such that

$$\widetilde{\epsilon} < \frac{\epsilon}{4 \exp(2\hat{t}M)}, \ \ \bar{\rho}(\hat{t})\widetilde{\epsilon} < \epsilon, \ \ \widetilde{\rho}(\hat{t})\widetilde{\epsilon} < \epsilon$$

where functions $\bar{\rho}(\cdot), \widetilde{\rho}(\cdot)$ are defined in Corollary J.6 and J.9 respectively.

Let us define an event $A^\times = A_1^\times \cup A_2^\times$ where

$$A_1^\times \triangleq \Big\{\exists n < T_1^\eta(\delta) \text{ s.t. } \|X_n^\eta(x)\| > 3\epsilon\Big\},$$

$$A_2^\times \triangleq \Big\{\exists j = 2, \cdots, l^* \text{ s.t.}$$

$$\max_{k=1,2,\cdots,(T_j^\eta(\delta)-T_{j-1}^\eta(\delta)-1)\wedge(2\hat{t}/\eta)} \eta \left\| Z_{T_{j-1}^\eta(\delta)+1} + \cdots + Z_{T_{j-1}^\eta(\delta)+j} \right\| > \widetilde{\epsilon}\Big\}$$

where the positive integer $l^*$ is defined in eq. (J.1). By definition of $l^*$, we must have $K^\eta(\delta) < l^*$ on event $\{\mathcal{J}_{\eta,\delta}^\downarrow(\sigma(\eta)) < J_\mathcal{G}\}$.

Now let us analyze the dynamics of $X_n^\eta$ on event $(A^\times)^c \cap \{\mathcal{J}_{\eta,\delta}^\downarrow(\sigma(\eta)) < J_\mathcal{G}\}$. In particular, we decompose it into two events

$$B_1 \triangleq (A^\times)^c \cap \{\mathcal{J}_{\eta,\delta}^\downarrow(\sigma(\eta)) < J_\mathcal{G}\}$$
$$\cap \{\forall j = 1, 2, \cdots, K^\eta(\delta), \ [T_{j+1}^\eta(\delta) \wedge \sigma(\eta) \wedge R(\epsilon, \delta, \eta)] - T_j^\eta(\delta) \leq 2\hat{t}/\eta\},$$
$$B_2 \triangleq (A^\times)^c \cap \{\mathcal{J}_{\eta,\delta}^\downarrow(\sigma(\eta)) < J_\mathcal{G}\}$$
$$\cap \{\exists j = 1, 2, \cdots, K^\eta(\delta) \text{ s.t. } [T_{j+1}^\eta(\delta) \wedge \sigma(\eta) \wedge R(\epsilon, \delta, \eta)] - T_j^\eta(\delta) > 2\hat{t}/\eta\}.$$

Using Lemma J.15 and the fact that $B_1 = \big(\bigcup_{K=1}^{l^*} B_1^\times(K) \cap \{K^\eta(\delta) = K\}\big) \cap \{\mathcal{J}_{K^\eta(\delta)}^{\eta,\delta} < J_\mathcal{G}\}$, one can see that for any $\eta > 0$ sufficiently small, we must have

$$\boldsymbol{d}(X_{\sigma(\eta)\wedge R(\epsilon,\delta,\eta)}^\eta(x), \mathcal{G}^c) > 100 l^* \bar{\epsilon} - 2\epsilon > 100 l^* \bar{\epsilon} - \bar{\epsilon} > 0.$$

on event $B_1$. In other words, on event $B_1$ we must have $R(\epsilon, \delta, \eta) < \sigma(\eta)$.

On the other hand, Lemma J.16 the fact that $B_2 \subseteq \bigcup_{K=1}^{l^*} B_2^\times(K) \cap \{K^\eta(\delta) = K\}$ implies that $B_2 = \emptyset$ whenever $\eta > 0$ is sufficiently small.

In summary, we have established that

$$\sup_{x: \|x\| \le 2\epsilon} \mathbb{P}_x \Big( \sigma(\eta) < R(\epsilon, \eta), \ \mathcal{J}_{\eta,\delta}^\downarrow(\sigma(\eta)) < J_\mathcal{G} \Big) \le \sup_{x: \|x\| \le 2\epsilon} \mathbb{P}(A^\times).$$

Lastly, from Lemma J.10 and J.12, one can see that for any $\delta > 0$ that is sufficiently small,

$$\sup_{x: \|x\| \le 2\epsilon} \mathbb{P}(A^\times) = o(\eta^N),$$

and this concludes the proof. $\qquad\square$

Using an almost identical approach, we can establish the next two results and conclude that it is also rather unlikely to have scenarios where

- The accumulated cost of large jumps exceeds $J_\mathcal{G}$ before the first return or exit,
- Or the first return occurs before the first exit,

yet it takes rather long for the said event to occur.

**Lemma J.18.** *Given $\epsilon \in (0, \bar{\epsilon}/3)$, $N > 0$, the following claim holds for any sufficiently small $\delta > 0$:*

$$\sup_{x \in \overline{B(\mathbf{0}, 2\epsilon)}} \mathbb{P}_x \Big( \exists K \in \mathbb{N} \ s.t. \ \mathcal{J}_K^{\eta,\delta} \ge J_\mathcal{G} \ and \ T_j^\eta(\delta) - T_{j-1}^\eta(\delta) > 2\hat{t}(\epsilon)/\eta \ for \ some \ j = 2, 3, \cdots, K \Big)$$

$$= o(\eta^N)$$

*as $\eta \downarrow 0$ where $\hat{t}(\epsilon) = c_1 + c_1 \log(1/\epsilon)$.*

*Proof.* Given the fixed $\epsilon > 0$, define $\hat{t} = c_1 + c_1 \log(1/\epsilon)$. Also, fix $\tilde{\epsilon} > 0$ as the largest possible value such that

$$\tilde{\epsilon} \le \frac{\epsilon}{4 \exp(2\hat{t}M)}, \ \ \bar{\rho}(\hat{t})\tilde{\epsilon} \le \epsilon, \ \ \tilde{\rho}(\hat{t})\tilde{\epsilon} \le \epsilon$$

where functions $\bar{\rho}(\cdot), \tilde{\rho}(\cdot)$ are defined in Corollary J.6 and J.9 respectively.

Let us define an event $A^\times = A_1^\times \cup A_2^\times$ where

$$A_1^\times \triangleq \Big\{ \exists n < T_1^\eta(\delta) \ s.t. \ \|X_n^\eta(x)\| > 3\epsilon \Big\},$$

$$A_2^\times \triangleq \Big\{ \exists j = 2, \cdots, l^* \ s.t.$$

$$\max_{k=1,2,\cdots,(T_j^\eta(\delta)-T_{j-1}^\eta(\delta)-1)\wedge(2\hat{t}/\eta)} \eta \Big\| Z_{T_{j-1}^\eta(\delta)+1} + \cdots + Z_{T_{j-1}^\eta(\delta)+j} \Big\| > \tilde{\epsilon} \Big\}$$

where the positive integer $l^*$ is defined in eq. (J.1). Furthermore, define event

$$B \triangleq (A^\times)^c \cap \Big\{ \exists K \in \mathbb{N} \text{ s.t. } \mathcal{J}_K^{\eta,\delta} \ge J_\mathcal{G} \text{ and } T_j^\eta(\delta) - T_{j-1}^\eta(\delta) > \frac{2\hat{t}}{\eta} \text{ for some } j = 2, 3, \cdots, K \Big\}.$$

Note that on event $B$, the index

$$K^* = \max\{k \ge 1 : \mathcal{J}_j^{\eta,\delta} < J_\mathcal{G} \ \forall j = 1, 2, \cdots, k-1\}$$

is well defined with $1 \le K^* < l^*$ (due to the definition of $l^*$). As a consequence, we must have $T_{K^*}^\eta(\delta) \le \sigma(\eta) \wedge R(\epsilon, \delta, \eta)$.

Now based on the exact value of $K^*$, we can decompose the event $B$ into $B = \cup_{K=0}^{l^*} B(K)$ where

$$B(K) \triangleq (A^\times)^c \cap \{K^* - 1 = K\} \cap \{\mathcal{J}_K^{\eta,\delta} < J_\mathcal{G}\} \cap \{T_{j+1}^\eta(\delta) - T_j^\eta(\delta) > \frac{2\hat{t}}{\eta} \text{ for some } j \in [K]\}.$$

By applying Lemma J.16, we know that for all $\eta > 0$ sufficiently small, $B(K) = \emptyset$. Therefore, for all sufficiently small $\eta$,

$$\sup_{x \in \overline{B(\mathbf{0}, 2\epsilon)}} \mathbb{P}_x \Big( \exists K \in \mathbb{N} \text{ s.t. } \mathcal{J}_K^{\eta, \delta} \geq J_{\mathcal{G}} \text{ and } T_j^\eta(\delta) - T_{j-1}^\eta(\delta) > 2\hat{t}(\epsilon)/\eta \text{ for some } j = 2, 3, \cdots, K \Big)$$

$$\leq \sup_{x \in \overline{B(\mathbf{0}, 2\epsilon)}} \mathbb{P}_x(A^\times).$$

Lastly, from Lemma J.10 and J.12, one can see that for any $\delta > 0$ that is sufficiently small,

$$\sup_{x: \; \|x\| \leq 2\epsilon} \mathbb{P}(A^\times) = o(\eta^N),$$

and this concludes the proof. $\qquad\qquad\qquad\qquad\qquad\qquad\qquad\qquad\qquad\qquad\qquad\qquad\square$

**Lemma J.19.** *Given $\epsilon \in (0, \bar{\epsilon}/3)$, $N > 0$, the following claim holds for any sufficiently small $\delta > 0$:*

$$\sup_{x \in \overline{B(\mathbf{0}, 2\epsilon)}} \mathbb{P}_x \Big( R(\epsilon, \delta, \eta) < \sigma(\eta), \; J_{K^\eta(\delta)}^{\eta, \delta} < J_{\mathcal{G}},$$

$$T_{j+1}^\eta(\delta) \wedge R(\epsilon, \delta, \eta) - T_j^\eta(\delta) > \frac{2\hat{t}(\epsilon)}{\eta} \text{ for some } j \in [K^\eta(\delta)] \Big) = o(\eta^N) \quad \text{as } \eta \downarrow 0$$

*where $\hat{t}(\epsilon) = c_1 + c_1 \log(1/\epsilon)$.*

*Proof.* Given the fixed $\epsilon > 0$, define $\hat{t} = c_1 + c_1 \log(1/\epsilon)$. Also, fix $\widetilde{\epsilon} > 0$ as the largest possible value such that

$$\widetilde{\epsilon} \leq \frac{\epsilon}{4 \exp(2\hat{t}M)}, \quad \bar{\rho}(\hat{t})\widetilde{\epsilon} \leq \epsilon, \quad \widetilde{\rho}(\hat{t})\widetilde{\epsilon} \leq \epsilon$$

where functions $\bar{\rho}(\cdot), \widetilde{\rho}(\cdot)$ are defined in Corollary J.6 and J.9 respectively.

Let us define an event $A^\times = A_1^\times \cup A_2^\times$ where

$$A_1^\times \triangleq \Big\{ \exists n < T_1^\eta(\delta) \text{ s.t. } \|X_n^\eta(x)\| > 3\epsilon \Big\},$$

$$A_2^\times \triangleq \Big\{ \exists j = 2, \cdots, l^* \text{ s.t.}$$

$$\max_{k = 1, 2, \cdots, (T_j^\eta(\delta) - T_{j-1}^\eta(\delta) - 1) \wedge (2\hat{t}/\eta)} \eta \left\| Z_{T_{j-1}^\eta(\delta)+1} + \cdots + Z_{T_{j-1}^\eta(\delta)+j} \right\| > \widetilde{\epsilon} \Big\}$$

where the positive integer $l^*$ is defined in eq. (J.1). By definition of $l^*$, we must have $K^\eta(\delta) < l^*$ on event $\{J_{K^\eta(\delta)}^{\eta, \delta} < J_{\mathcal{G}}\}$. Furthermore, define event

$$B \triangleq (A^\times)^c \cap \Big\{ R(\epsilon, \delta, \eta) < \sigma(\eta), \; J_{K^\eta(\delta)}^{\eta, \delta} < J_{\mathcal{G}},$$

$$T_{j+1}^\eta(\delta) \wedge R(\epsilon, \delta, \eta) - T_j^\eta(\delta) > \frac{2\hat{t}(\epsilon)}{\eta} \text{ for some } j \in [K^\eta(\delta)] \Big\}.$$

Now observe that $B \subseteq \bigcup_{K=1}^{l^*} B(K)$ where

$$B(K) \triangleq (A^\times)^c \cap \{K^\eta(\delta) = K\} \cap \{\mathcal{J}_K^{\eta, \delta} < J_{\mathcal{G}}\}$$

$$\cap \{\exists j \in [K^\eta(\delta)] \text{ s.t. } [T_{j+1}^\eta(\delta) \wedge \sigma(\eta) \wedge R(\epsilon, \delta, \eta)] - T_j^\eta(\delta) > \frac{2\hat{t}}{\eta}\}.$$

By applying Lemma J.16, we know that for all $\eta > 0$ sufficiently small, $B(K) = \emptyset$. Therefore, for all sufficiently small $\eta$,

$$\sup_{x \in \overline{B(\mathbf{0}, 2\epsilon)}} \mathbb{P}_x \Big( R(\epsilon, \delta, \eta) < \sigma(\eta), \; J_{K^\eta(\delta)}^{\eta, \delta} < J_{\mathcal{G}},$$

$$T_{j+1}^\eta(\delta) \wedge R(\epsilon, \delta, \eta) - T_j^\eta(\delta) > \frac{2\hat{t}(\epsilon)}{\eta} \text{ for some } j \in [K^\eta(\delta)] \Big) \leq \sup_{x \in \overline{B(\mathbf{0}, 2\epsilon)}} \mathbb{P}_x(A^\times).$$

Lastly, from Lemma J.10 and J.12, one can see that for any $\delta > 0$ that is sufficiently small,

$$\sup_{x:\ \|x\| \leq 2\epsilon} \mathbb{P}(A^\times) = o(\eta^N),$$

and this concludes the proof. $\qquad\square$

Recall that in Lemma J.17, we have shown that it is rather unlikely to have the first exit with accumulated cost of large jumps less than $J_{\mathcal{G}}$. In the next result, we show that even with large jumps of accumulated cost $J_{\mathcal{G}}$, if some jumps are still not large enough, or the inter-arrival times are too long, then it is still very unlikely for the SGD iterates to even get close the the boundary. Define

$$K_{J_{\mathcal{G}}}^{\eta,\delta} \triangleq \min\{k \in [K^\eta(\delta)] :\ \mathcal{J}_k^{\eta,\delta} \geq J_{\mathcal{G}}\}$$

and let $K_{J_{\mathcal{G}}}^{\eta,\delta} = \infty$ if $\mathcal{J}_{K^\eta(\delta)}^{\eta,\delta} < J_{\mathcal{G}}$. Also, if $K_{J_{\mathcal{G}}}^{\eta,\delta} < \infty$, let

$$T^{\geq J_{\mathcal{G}}}(\eta,\delta) \triangleq T_{K_{J_{\mathcal{G}}}^{\eta,\delta}}^\eta(\delta)$$

In other words, $T^{\geq J_{\mathcal{G}}}(\eta,\delta)$ is the first time that, prior to first exit or return, the accumulated cost of large jumps has reached $J_{\mathcal{G}}$. Regarding the $o(\eta^{1+J_{\mathcal{G}}-\alpha_1+\Delta})$ term in the next Lemma, we note that the function $\widetilde{\lambda}$ defined in eq. (J.50) is regularly varying (as $\eta \downarrow 0$) with index $1 + J_{\mathcal{G}} - \alpha_1$. Therefore, $\eta^{1+J_{\mathcal{G}}-\alpha_1+\Delta} = o(\widetilde{\lambda}(\eta))$ as $\eta$ approaches 0.

**Lemma J.20.** *There exists some $\Delta > 0$ such that the following claim holds for all $\epsilon \in (0, \bar{\epsilon}/3)$ and all $\delta > 0$ that is sufficiently small:*

$$\sup_{x:\ \|x\| \leq 2\epsilon} \mathbb{P}_x\big(B_{2,(I)}^\times(\epsilon,\delta,\eta) \cup B_{2,(II)}^\times(\epsilon,\delta,\eta)\big) = o(\eta^{1+J_{\mathcal{G}}-\alpha_1+\Delta})$$

*as $\eta \downarrow 0$ where $\hat{t}(\epsilon) = c_1 + c_1 \log(1/\epsilon)$ and*

$$B_{2,(I)}^\times(\epsilon,\delta,\eta) = \{K_{J_{\mathcal{G}}}^{\eta,\delta} < \infty\} \cap \{\mathcal{J}_{K_{J_{\mathcal{G}}}^{\eta,\delta}}^{\eta,\delta} > J_{\mathcal{G}}\},$$

$$B_{2,(II)}^\times(\epsilon,\delta,\eta) = \{K_{J_{\mathcal{G}}}^{\eta,\delta} < \infty\} \cap \{\mathcal{J}_{K_{J_{\mathcal{G}}}^{\eta,\delta}}^{\eta,\delta} = J_{\mathcal{G}}\}$$

$$\cap \left\{\exists j = 2,3,\cdots, K_{J_{\mathcal{G}}}^{\eta,\delta}\ s.t. T_j^\eta(\delta) - T_{j-1}^\eta(\delta) > 2\bar{t}/\eta\ or\ \exists j = 1,2,\cdots, K_{J_{\mathcal{G}}}^{\eta,\delta}\ s.t.\ \eta \big\|W_j^\eta(\delta)\big\| \leq \bar{\delta}\right\}$$

$$\cap \left\{\min_{j=0,1,\cdots, T^{\geq J_{\mathcal{G}}}(\eta,\delta)} d\big(X_j^\eta, \mathcal{G}^c\big) \leq \bar{\epsilon}\right\}.$$

*Proof.* From Assumption I.8, we know the existence of some $\Delta > 0$ satisfying the following condition: For any $k \in \mathbb{N}$ and any $\boldsymbol{j} \in \{1,\cdots,m\}^k$ such that $\sum_{i=1}^{k-1}(\alpha_{j_i}-1) < J_{\mathcal{G}} < \sum_{i=1}^{k}(\alpha_{j_i}-1)$ (note that there are only finitely many possible choices for such $\boldsymbol{j}$), we have

$$J_{\mathcal{G}} + \Delta < \sum_{i=1}^{k}(\alpha_{j_i}-1) \tag{J.55}$$

Fix such $\Delta > 0$. Meanwhile, given the fixed $\epsilon > 0$, we can define $\hat{t} = c_1 + c_1 \log(1/\epsilon)$ and choose $\widetilde{\epsilon} > 0$ as the largest possible value such that

$$\widetilde{\epsilon} \leq \frac{\epsilon}{4\exp(2\hat{t}M)},\quad \bar{\rho}(\hat{t})\widetilde{\epsilon} \leq \epsilon,\quad \widetilde{\rho}(\hat{t})\widetilde{\epsilon} \leq \epsilon$$

where functions $\bar{\rho}(\cdot), \widetilde{\rho}(\cdot)$ are defined in Corollary J.6 and J.9 respectively. We stress that $\bar{t}$ is a fixed constant while $\hat{t}$ depends on the value of $\epsilon$, and for any sufficiently small $\epsilon$ we will have $\hat{t} > \bar{t}$.

Let us define an event $A^\times = A_1^\times \cup A_2^\times$ where

$$A_1^\times \triangleq \left\{\exists n < T_1^\eta(\delta)\ s.t.\ \|X_n^\eta(x)\| > 3\epsilon\right\},$$

$$A_2^\times \triangleq \Big\{\exists j = 2,\cdots,l^*\ s.t.$$

$$\max_{k=1,2,\cdots,(T_j^\eta(\delta)-T_{j-1}^\eta(\delta)-1)\wedge(2\hat{t}/\eta)} \eta \left\|Z_{T_{j-1}^\eta(\delta)+1} + \cdots + Z_{T_{j-1}^\eta(\delta)+j}\right\| > \widetilde{\epsilon}\Big\}$$

where the positive integer $l^*$ is defined in eq. (J.1). By definition of $l^*$, we must have $K_{J_{\mathcal{G}}}^{\eta,\delta} < l^*$ on event $\{K_{J_{\mathcal{G}}}^{\eta,\delta} < \infty\}$.

First, we analyze the following event

$$B \triangleq (A^\times)^c \cap \{K_{J_{\mathcal{G}}}^{\eta,\delta} < \infty\} \cap \Big\{ T_j^\eta(\delta) - T_{j-1}^\eta(\delta) > \frac{2\hat{t}}{\eta} \text{ for some } j = 2, 3, \cdots, K_{J_{\mathcal{G}}}^{\eta,\delta} \Big\}.$$

In particular, note that $B = \bigcup_{K=0}^{l^*-1} B(K)$ where

$$B(K) \triangleq (A^\times)^c \cap \{K_{J_{\mathcal{G}}}^{\eta,\delta} = K+1\} \cap \{\mathcal{J}_K^{\eta,\delta} < J_{\mathcal{G}}\}$$

$$\cap \Big\{ T_{j+1}^\eta(\delta) - T_j^\eta(\delta) > \frac{2\hat{t}}{\eta} \text{ for some } j = 1, 2, 3, \cdots, K \Big\}.$$

When $K = 0 < 1$, the event $B(0) = \emptyset$ by definition (due to event $\Big\{ T_{j+1}^\eta(\delta) - T_j^\eta(\delta) > \frac{2\hat{t}}{\eta} \text{ for some } j = 1, 2, 3, \cdots, K \Big\}$). For $K = 1, \cdots, l^*$, Lemma J.16 implies that $B(K) = \emptyset$ for all $\eta$ sufficiently small. In summary, $B = \emptyset$ for all $\eta$ sufficiently small.

Now we focus on the following two events

$$C \triangleq (A^\times)^c \cap \{K_{J_{\mathcal{G}}}^{\eta,\delta} < \infty\} \cap \Big\{ T_j^\eta(\delta) - T_{j-1}^\eta(\delta) \leq \frac{2\hat{t}}{\eta} \text{ for all } j = 2, 3, \cdots, K_{J_{\mathcal{G}}}^{\eta,\delta} \Big\} \cap \{\mathcal{J}_{K^\eta(\delta)}^{\eta,\delta} > J_{\mathcal{G}}\},$$

$$D \triangleq (A^\times)^c \cap \{K_{J_{\mathcal{G}}}^{\eta,\delta} < \infty\} \cap \Big\{ T_j^\eta(\delta) - T_{j-1}^\eta(\delta) \leq \frac{2\hat{t}}{\eta} \text{ for all } j = 2, 3, \cdots, K_{J_{\mathcal{G}}}^{\eta,\delta} \Big\} \cap \{\mathcal{J}_{K^\eta(\delta)}^{\eta,\delta} = J_{\mathcal{G}}\}.$$

On the one hand, event $C$ can be decomposed as follows. Let

$$\mathbb{J}^\uparrow \triangleq \{\boldsymbol{j} \in \{1, 2, \cdots, m\}^k : \ k = |\boldsymbol{j}|, \ \sum_{i=1}^{k-1}(\alpha_{j_i} - 1) < J_{\mathcal{G}} < \sum_{i=1}^{k}(\alpha_{j_i} - 1)\}$$

be the set that contains all the types $\boldsymbol{j}$ such that the accumulated cost reach $J_{\mathcal{G}}$ if and only if the last element is kept. One can see that there are only finitely many elements in $\mathbb{J}^\uparrow$. Now we have $C = \bigcup_{\boldsymbol{j} \in \mathbb{J}^\uparrow} C(\boldsymbol{j})$ where (see the definition in eq. (J.10))

$$C(\boldsymbol{j}) \triangleq (A^\times)^c \cap \{K_{J_{\mathcal{G}}}^{\eta,\delta} < \infty\} \cap \Big\{ T_j^\eta(\delta) - T_{j-1}^\eta(\delta) \leq \frac{2\hat{t}}{\eta} \text{ for all } j = 2, 3, \cdots, K_{J_{\mathcal{G}}}^{\eta,\delta} \Big\}$$

$$\cap \Big\{ \big( W_1^\eta(\delta), W_2^\eta(\delta), \cdots, W_{K_{J_{\mathcal{G}}}^{\eta,\delta}}^\eta(\delta) \big) \text{ is of type-}\boldsymbol{j} \Big\}.$$

For any $\boldsymbol{j} \in \mathbb{J}^\uparrow$, observe that (let $k_{\boldsymbol{j}} = |\boldsymbol{j}|$)

$$\sup_{\|x\| \leq 2\epsilon} \mathbb{P}_x(C(\boldsymbol{j}))$$

$$\leq \mathbb{P}\Big( W_i^\eta(\delta) \in E_{\boldsymbol{j}_i} \ \forall i \in [K_{J_{\mathcal{G}}}^{\eta,\delta}] \text{ and } T_i^\eta(\delta) - T_{i-1}^\eta(\delta) \leq \frac{2\hat{t}}{\eta} \text{ for all } i = 2, 3, \cdots, K_{J_{\mathcal{G}}}^{\eta,\delta} \Big)$$

$$= \Big( \prod_{i=1}^{k_{\boldsymbol{j}}} \frac{H_{\boldsymbol{j}_i}(\delta/\eta)}{H(\delta/\eta)} \Big) \cdot \big( \mathbb{P}(T_1^\eta(\delta) \leq 2\hat{t}/\eta) \big)^{k_{\boldsymbol{j}}-1}.$$

The last equality is due to independence of $W_i^\eta(\delta)$ and $T_i^\eta(\delta) - T_{i-1}^\eta(\delta)$. The regularly varying natures of functions $H_j$ and $H$ imply that

$$\lim_{\eta \downarrow 0} \frac{\prod_{i=1}^{k_{\boldsymbol{j}}} \frac{H_{\boldsymbol{j}_i}(\delta/\eta)}{H(\delta/\eta)}}{\prod_{i=1}^{k_{\boldsymbol{j}}} \frac{H_{\boldsymbol{j}_i}(1/\eta)}{H(1/\eta)}} = \frac{\delta^{k_{\boldsymbol{j}}\alpha_1}}{\delta^{\sum_{i=1}^{k_{\boldsymbol{j}}} \alpha_{\boldsymbol{j}_i}}}.$$

Meanwhile, from Lemma G.4 and the regularly varying nature of function $H_j$ and $H$ (in particular, repeating the same calculations that leads to eq. (J.46)), we have

$$\lim_{\eta \downarrow 0} \frac{\big( \mathbb{P}(T_1^\eta(\delta) \leq 2\hat{t}/\eta) \big)^{k_{\boldsymbol{j}}-1}}{\big( \frac{H(1/\eta)}{\eta} \cdot \frac{2\hat{t}}{\delta^{\alpha_1}} \big)^{k_{\boldsymbol{j}}-1}} = 1.$$

Therefore, we have yielded that

$$\limsup_{\eta \downarrow 0} \frac{\sup_{\|x\| \leq 2\epsilon} \mathbb{P}_x(C(\boldsymbol{j}))}{\widetilde{\lambda}_{\boldsymbol{j}}(\eta)} \leq \frac{\delta^{k_j \alpha_1}}{\delta^{\sum_{i=1}^{k_j} \alpha_{j_i}}} \cdot \Big(\frac{2\hat{t}}{\delta^{\alpha_1}}\Big)^{k_j - 1} < \infty$$

where

$$\widetilde{\lambda}_{\boldsymbol{j}}(\eta) \triangleq \frac{\prod_{i=1}^{k_j} H_{\boldsymbol{j}_i}(1/\eta)}{\eta^{k_j - 1} H(1/\eta)} = o(\eta^{1 + J_{\mathcal{G}} - \alpha_1 + \Delta})$$

due to eq. (J.55). Therefore, one can see that as $\eta \downarrow 0$,

$$\sup_{\|x\| \leq 2\epsilon} \mathbb{P}_x(C) = o(\eta^{1 + J_{\mathcal{G}} - \alpha_1 + \Delta}).$$

Furthermore, the discussion above have shown that for all $\eta$ sufficiently small,

$$\sup_{x:\ \|x\| \leq 2\epsilon} \mathbb{P}_x\big(B_{2,\mathrm{(I)}}^{\times}(\epsilon, \delta, \eta)\big) \leq o(\eta^{1 + J_{\mathcal{G}} - \alpha_1 + \Delta}) + \sup_{x:\ \|x\| \leq 2\epsilon} \mathbb{P}_x(A^{\times}).$$

From Lemma J.10 and J.12, one can see that for all $N > 1 + J_{\mathcal{G}} - \alpha_1 + \Delta$ and all $\delta > 0$ that is sufficiently small, $\sup_{x:\ \|x\| \leq 2\epsilon} \mathbb{P}(A^{\times}) = o(\eta^N)$. Fix such $\delta > 0$. In conclusion, we now have that $\sup_{x:\ \|x\| \leq 2\epsilon} \mathbb{P}_x\big(B_{2,\mathrm{(I)}}^{\times}(\epsilon, \delta, \eta)\big) = o(\eta^{1 + J_{\mathcal{G}} - \alpha_1 + \Delta})$.

On the other hand, on event $D$, Assumption I.7 shows that (see the definition in eq. (J.39))

$$\{K_{J_{\mathcal{G}}}^{\eta, \delta} < \infty\} \cap \{\mathcal{J}_{K^{\eta}(\delta)}^{\eta, \delta} = J_{\mathcal{G}}\}$$
$$= \{K_{J_{\mathcal{G}}}^{\eta, \delta} < \infty\} \cap \big\{ \big(W_1^{\eta}(\delta), W_2^{\eta}(\delta), \cdots, W_{K_{J_{\mathcal{G}}}^{\eta, \delta}}^{\eta}(\delta)\big) \text{ is of type-}\boldsymbol{j} \text{ for some } \boldsymbol{j} \in \boldsymbol{j}(\boldsymbol{i}^*)\big\}.$$

Therefore,

$$B_{2,\mathrm{(II)}}^{\times}(\epsilon, \delta, \eta) \cap D$$
$$= (A^{\times})^c \cap \{K_{J_{\mathcal{G}}}^{\eta, \delta} < \infty\} \cap \big\{ \big(W_1^{\eta}(\delta), W_2^{\eta}(\delta), \cdots, W_{K_{J_{\mathcal{G}}}^{\eta, \delta}}^{\eta}(\delta)\big) \text{ is of type-}\boldsymbol{j} \text{ for some } \boldsymbol{j} \in \boldsymbol{j}(\boldsymbol{i}^*)\big\}$$
$$\cap \Big\{\forall j = 2, 3, \cdots, K_{J_{\mathcal{G}}}^{\eta, \delta},\ T_j^{\eta}(\delta) - T_{j-1}^{\eta}(\delta) \leq 2\hat{t}/\eta\Big\} \cap \Big\{ \min_{j=0,1,\cdots,T^{\geq J_{\mathcal{G}}}(\eta, \delta)} \boldsymbol{d}\big(X_j^{\eta}, \mathcal{G}^c\big) \leq \bar{\epsilon}\Big\}$$
$$\cap \Big\{\exists j = 2, 3, \cdots, K_{J_{\mathcal{G}}}^{\eta, \delta} \text{ s.t. } T_j^{\eta}(\delta) - T_{j-1}^{\eta}(\delta) > 2\bar{t}/\eta \text{ OR } \exists j = 1, 2, \cdots, K_{J_{\mathcal{G}}}^{\eta, \delta} \text{ s.t. } \eta \big\|W_j^{\eta}(\delta)\big\| \leq \bar{\delta}\Big\}.$$

In particular, using Lemma J.15, one can see that when $\eta$ is sufficiently small, on event $B_{2,\mathrm{(II)}}^{\times}(\epsilon, \delta, \eta) \cap D$ we have $\|X_k^{\eta}\| < 3\epsilon$ for all $k < T_1^{\eta}(\delta)$ and

$$\sup_{s \in [0, T^{\geq J_{\mathcal{G}}}(\eta, \delta) - T_1^{\eta}(\delta)]} \Big\|X_{\lfloor s \rfloor}^{\eta} - \boldsymbol{z}^{\eta}(s)\Big\| < 2\epsilon.$$

Here

$$\boldsymbol{z}^{\eta}(s) = \widetilde{\boldsymbol{x}}^{\eta}\big(s, X^{(1)}; \boldsymbol{T}, \boldsymbol{W}\big) \ \forall s \geq 0$$

with $\boldsymbol{T} = \big(0, T_2^{\eta}(\delta) - T_1^{\eta}(\delta), T_3^{\eta}(\delta) - T_2^{\eta}(\delta), \cdots, T_{K_{J_{\mathcal{G}}}^{\eta, \delta} + 1}^{\eta}(\delta) - T_{K_{J_{\mathcal{G}}}^{\eta, \delta}}^{\eta}(\delta)\big)$, $\boldsymbol{W} = \big(W_1^{\eta}(\delta), W_2^{\eta}(\delta), \cdots, W_{K_{J_{\mathcal{G}}}^{\eta, \delta}}^{\eta}(\delta)\big)$, and $X^{(1)} = X_{T_1^{\eta}(\delta) - 1}^{\eta}$. Besides, eq. (J.17) dictates that, on event $B_{2,\mathrm{(II)}}^{\times}(\epsilon, \delta, \eta) \cap D$, we must have $\inf_{s \geq 0} \boldsymbol{d}(\boldsymbol{z}^{\eta}(s), \mathcal{G}^c) \geq 100 l^* \bar{\epsilon}$, hence

$$\min_{k \in [T^{\geq J_{\mathcal{G}}}(\eta, \delta)]} \boldsymbol{d}(X_k^{\eta}, \mathcal{G}^c) > 50 l^* \bar{\epsilon}.$$

However, this clearly contradicts the definition of event $B_{2,\mathrm{(II)}}^{\times}(\epsilon, \delta, \eta)$. In conclusion, we have established that (for all $\eta$ sufficiently small) $B_{2,\mathrm{(II)}}^{\times}(\epsilon, \delta, \eta) \cap D = \emptyset$ and

$$\sup_{x:\ \|x\| \leq 2\epsilon} \mathbb{P}_x\big(B_{2,\mathrm{(II)}}^{\times}(\epsilon, \delta, \eta)\big) \leq \sup_{x:\ \|x\| \leq 2\epsilon} \mathbb{P}_x(A^{\times}).$$

However, as per the argument above, our choice of sufficiently small $\delta > 0$ ensures that $\sup_{x:\ \|x\| \leq 2\epsilon} \mathbb{P}(A^{\times}) = o(\eta^{1 + J_{\mathcal{G}} - \alpha_1 + \Delta})$ and concludes the proof. $\qquad \square$

Furthermore, starting from the local minimum $\mathbf{0}$, it is unlikely that the SGD iterates will be extremely close to $\partial\mathcal{G}$ when the accumulated cost of large jumps reaches $J_{\mathcal{G}}$. The next lemma provides an upper bound for the probability of the said scenario. For any set $A \subseteq \mathbb{R}^d$, define its $\epsilon-$enlargement as the following open set $A^\epsilon \triangleq \{x: \ \boldsymbol{d}(x, A) < \epsilon\}$.

**Lemma J.21.** *For any $\epsilon \in \big(0, \bar{\epsilon}/(3 + 3\rho^*)\big)$ and any $\delta > 0$ that is sufficiently small,*

$$\limsup_{\eta \downarrow 0} \frac{\sup_{x: \ \|x\| \le 2\epsilon} \mathbb{P}_x(B_3^\times(\epsilon, \delta, \eta))}{\widetilde{\lambda}(\eta)} \le \delta^{\alpha_1} \Psi(\epsilon)$$

*where $B_3^\times(\epsilon, \delta, \eta) = \{K_{J_{\mathcal{G}}}^{\eta,\delta} < \infty\} \cap \{\boldsymbol{d}(X_{T \ge J_{\mathcal{G}}(\eta,\delta)}^\eta, \partial\mathcal{G}) < \epsilon\}$ and $\Psi(\epsilon) = \mu\Big(h^{-1}\big((\partial\mathcal{G})^{(3+3\rho^*)\epsilon}\big)\Big)$ with $h$ defined in eq. (J.38) and $\mu$ defined in eq. (J.48), and $\rho^* \in (0, \infty)$ is the constant provided in Corollary J.7.*

*Proof.* Again, given the fixed $\epsilon > 0$, we can define $\hat{t} = c_1 + c_1 \log(1/\epsilon)$ and choose $\widetilde{\epsilon} > 0$ as the largest possible value such that

$$\widetilde{\epsilon} \le \frac{\epsilon}{4 \exp(2\hat{t}M)}, \quad \bar{\rho}(\hat{t})\widetilde{\epsilon} \le \epsilon, \quad \widetilde{\rho}(\hat{t})\widetilde{\epsilon} \le \epsilon$$

where functions $\bar{\rho}(\cdot), \widetilde{\rho}(\cdot)$ are defined in Corollary J.6 and J.9 respectively. We stress that $\bar{t}$ is a fixed constant while $\hat{t}$ depends on the value of $\epsilon$, and for any sufficiently small $\epsilon$ we will have $\hat{t} > \bar{t}$.

Let us define an event $A^\times = A_1^\times \cup A_2^\times$ where

$$A_1^\times \triangleq \Big\{ \exists n < T_1^\eta(\delta) \ s.t. \ \|X_n^\eta(x)\| > 3\epsilon \Big\},$$

$$A_2^\times \triangleq \Big\{ \exists j = 2, \cdots, l^* \ s.t.$$

$$\max_{k=1,2,\cdots,(T_j^\eta(\delta) - T_{j-1}^\eta(\delta) - 1) \wedge (2\hat{t}/\eta)} \eta \Big\| Z_{T_{j-1}^\eta(\delta)+1} + \cdots + Z_{T_{j-1}^\eta(\delta)+j} \Big\| > \widetilde{\epsilon} \Big\}$$

where the positive integer $l^*$ is defined in eq. (J.1). Consider the following decomposition of event $B_3^\times(\epsilon, \delta, \eta) \cap (A^\times)^c$:

$$C_{(\mathrm{I})} \triangleq B_3^\times(\epsilon, \delta, \eta) \cap (A^\times)^c \cap \{\mathcal{J}_{K_{J_{\mathcal{G}}}^{\eta,\delta}}^{\eta,\delta} > J_{\mathcal{G}}\}.$$

$$C_{(\mathrm{II})} \triangleq B_3^\times(\epsilon, \delta, \eta) \cap (A^\times)^c \cap \{\mathcal{J}_{K_{J_{\mathcal{G}}}^{\eta,\delta}}^{\eta,\delta} = J_{\mathcal{G}}\} \cap \Big\{ \exists j = 2, 3, \cdots, K_{J_{\mathcal{G}}}^{\eta,\delta} \ s.t.$$

$$T_j^\eta(\delta) - T_{j-1}^\eta(\delta) > 2\bar{t}/\eta \ \text{or} \ \exists j = 1, 2, \cdots, K_{J_{\mathcal{G}}}^{\eta,\delta} \ s.t. \ \eta \big\| W_j^\eta(\delta) \big\| \le \bar{\delta} \Big\}$$

$$C_{(\mathrm{III})} \triangleq \big( B_3^\times(\epsilon, \delta, \eta) \cap (A^\times)^c \big) \backslash \big( C_{(\mathrm{I})} \cup C_{(\mathrm{II})} \big).$$

Using Lemma J.20, we know that for all $\delta > 0$ sufficiently small,

$$\limsup_{\eta \downarrow 0} \frac{\sup_{\|x\| \le 2\epsilon} \mathbb{P}_x\big(C_{(\mathrm{I})} \cup C_{(\mathrm{II})}\big)}{\widetilde{\lambda}(\eta)} = 0.$$

As a result,

$$\limsup_{\eta \downarrow 0} \frac{\sup_{\|x\| \le 2\epsilon} \mathbb{P}_x(B_3^\times(\epsilon, \delta, \eta))}{\widetilde{\lambda}(\eta)} \le \limsup_{\eta \downarrow 0} \frac{\sup_{\|x\| \le 2\epsilon} \mathbb{P}_x(A^\times)}{\widetilde{\lambda}(\eta)} + \limsup_{\eta \downarrow 0} \frac{\sup_{\|x\| \le 2\epsilon} \mathbb{P}_x(C_{(\mathrm{III})})}{\widetilde{\lambda}(\eta)}.$$

From Lemma J.10 and J.12, one can see that for all $N > 1 + J_{\mathcal{G}} - \alpha_1$ and all $\delta > 0$ that is sufficiently small, we have $\sup_{x: \ \|x\| \le 2\epsilon} \mathbb{P}(A^\times) = o(\eta^N) = o(\widetilde{\lambda}(\eta))$.

Moving on, we focus on bounding the probability of event $C_{(\mathrm{III})}$. In particular, note that

$$C_{(\mathrm{III})} = (A^\times)^c \cap \{K_{J_{\mathcal{G}}}^{\eta,\delta} < \infty\} \cap \{\mathcal{J}_{K_{J_{\mathcal{G}}}^{\eta,\delta}}^{\eta,\delta} = J_{\mathcal{G}}\} \cap \{\boldsymbol{d}(X_{T \ge J_{\mathcal{G}}(\eta,\delta)}^\eta, \partial\mathcal{G}) < \epsilon\}$$

$$\cap \Big\{ \forall j = 2, 3, \cdots, K_{J_{\mathcal{G}}}^{\eta,\delta}, \ T_j^\eta(\delta) - T_{j-1}^\eta(\delta) \le 2\bar{t}/\eta \ \text{and} \ \forall j = 1, 2, \cdots, K_{J_{\mathcal{G}}}^{\eta,\delta}, \ \eta \big\| W_j^\eta(\delta) \big\| \ge \bar{\delta} \Big\}.$$

By applying Lemma J.15 on event $C_{\text{(III)}}$, one can see that when $\eta$ is sufficiently small, on this event we have $\|X_k^\eta\| < 3\epsilon$ for all $k < T_1^\eta(\delta)$ and

$$\sup_{s \in [0, T^{\geq J_{\mathcal{G}}}(\eta, \delta) - T_1^\eta(\delta)]} \left\| X_{\lfloor s \rfloor + T_1^\eta(\delta)3}^\eta - z^\eta(s) \right\| < 2\epsilon.$$

Here

$$z^\eta(s) = \widetilde{x}^\eta\big(s, X^{(1)}; \, \boldsymbol{T}, \boldsymbol{W}\big) \ \forall s \geq 0$$

with $\boldsymbol{T} = \big(0, T_2^\eta(\delta) - T_1^\eta(\delta), T_3^\eta(\delta) - T_2^\eta(\delta), \cdots, T_{K_{J_{\mathcal{G}}}^{\eta,\delta}}^\eta(\delta) - T_{K_{J_{\mathcal{G}}-1}^{\eta,\delta}}^\eta(\delta)\big)$, $\boldsymbol{W} = \big(W_1^\eta(\delta), W_2^\eta(\delta), \cdots, W_{K_{J_{\mathcal{G}}}^{\eta,\delta}}^\eta(\delta)\big)$, and $X^{(1)} = X_{T_1^\eta(\delta)-1}^\eta$. To better control the location of $z^\eta(s)$, we also construct

$$z_0^\eta(s) = \widetilde{x}^\eta\big(s, \boldsymbol{0}; \, \boldsymbol{T}, \boldsymbol{W}\big) \ \forall s \geq 0$$

where the only difference is that we substitute the initial value $X^{(1)}$ with $\boldsymbol{0}$. Note that on event $C_{\text{(III)}}$, we have $\big\|X^{(1)}\big\| < 3\epsilon$. By applying Corollary J.7 (the condition about line segments contained in $\mathcal{G}$ is verified due to eq. (J.18)), we have the following bound on event $C_{\text{(III)}}$:

$$\sup_{s \in [0, \sum_{i=1}^{K_{J_{\mathbb{G}}}^{\eta,\delta}} \boldsymbol{T}_i]} \|z^\eta(s) - z_0^\eta(s)\| < 3\rho^*\epsilon.$$

Now recall Definition J.1. Combining all the bounds we have obtained so far, we see that, when $\eta$ is sufficiently small, on event $C_{\text{(III)}}$ we have

$$d\Big(z_0^\eta\big(\sum_{i=1}^{K_{J_{\mathbb{G}}}^{\eta,\delta}} \boldsymbol{T}_i\big), \, \partial\mathcal{G}\Big) < (2 + 3\rho^*)\epsilon$$

and $\boldsymbol{W}$ is of type-$\boldsymbol{j}$ for some $\boldsymbol{j} \in \boldsymbol{j}(\boldsymbol{i}^*)$. It then follows immediately from Corollary J.14 that

$$\limsup_{\eta \downarrow 0} \frac{\sup_{\|x\| \leq 2\epsilon} \mathbb{P}_x(C_{\text{(III)}})}{\widetilde{\lambda}(\eta)} \leq \limsup_\eta \frac{p\big((\partial\mathcal{G})^{(2+3\rho^*)\epsilon}, \delta, \eta\big)}{\widetilde{\lambda}(\eta)} \leq \delta^{\alpha_1} \mu\Big(h^{-1}\big(\overline{(\partial\mathcal{G})^{(2+3\rho^*)\epsilon}}\big)\Big)$$

$$\leq \delta^{\alpha_1} \mu\Big(h^{-1}\big((\partial\mathcal{G})^{(3+3\rho^*)\epsilon}\big)\Big)$$

and this concludes the proof. $\qquad\square$

Lastly, we provide a lower bound for the probability of the *most likely* way of escape, i.e., due to $k^*$ large jumps with accumulated cost $J_{\mathcal{G}}$ and relatively short inter-arrival time less (when compared to $\bar{t}/\eta$). We stress that, in the next result, the value of constant $c_* > 0$ would not vary with the choice of parameters $\epsilon, \delta$.

**Lemma J.22.** *There exists some $c_* > 0$ such that the following claim holds for any $\epsilon \in (0, \bar{\epsilon}/(4 + 3\rho^*))$ and any sufficiently small $\delta > 0$:*

$$\liminf_{\eta \downarrow 0} \frac{\inf_{\|x\| \leq 2\epsilon} \mathbb{P}_x(A^\circ(\epsilon, \delta, \eta))}{\widetilde{\lambda}(\eta)} \geq c_* \delta^{\alpha_1}$$

*where the event is defined as*

$$A^\circ(\epsilon, \delta, \eta) \triangleq \{K_{J_{\mathcal{G}}}^{\eta,\delta} < \infty\} \cap \Big\{\sigma(\eta) = T^{\geq J_{\mathcal{G}}}(\eta, \delta)\Big\} \cap \Big\{T_j^\eta(\delta) - T_{j-1}^\eta(\delta) \leq \frac{2\bar{t}}{\eta} \ \forall j = 2, 3, \cdots, k^*\Big\}$$

*and $\rho^* \in (0, \infty)$ is the constant provided in Corollary J.7.*

*Proof.* First of all, Assumption I.7 implies that

$$\{K_{J_{\mathcal{G}}}^{\eta,\delta} < \infty\} \cap \Big\{\sigma(\eta) = T^{\geq J_{\mathcal{G}}}(\eta, \delta)\Big\} = \{K_{J_{\mathcal{G}}}^{\eta,\delta} = k^*\} \cap \Big\{\sigma(\eta) = T_{k^*}^\eta(\delta)\Big\}.$$

We also stress that $K_{J_{\mathcal{G}}}^{\eta,\delta} = k^*$ means that $\sigma(\eta) \wedge R(\epsilon, \delta, \eta) \geq T_{k^*}^{\eta}(\delta)$ (i.e. the arrival time of the $k^*$–th large jumps and that the accumulated cost of large jumps $W_1^{\eta}(\delta), \cdots, W_{k^*}^{\eta}(\delta)$ is exactly $J_{\mathcal{G}}$.

Due to eq. (J.19) and eq. (J.17), there exists some $\boldsymbol{j} \in \boldsymbol{j}(\boldsymbol{i}^*)$, some $\widetilde{\boldsymbol{\theta}} = (\widetilde{\theta}_1, \cdots, \widetilde{\theta}_{k^*})$ with $\widetilde{\theta}_i \in F_i \,\forall i$, some $\widetilde{\boldsymbol{r}} = (\widetilde{r}_1, \cdots, \widetilde{r}_{k^*})$ such that $\widetilde{r}_i \geq \bar{\delta} \,\forall i$, some $\widetilde{\boldsymbol{t}} = (\widetilde{t}_1, \cdots, \widetilde{t}_{k^*})$ such that $\widetilde{t}_1 = 0$ and $\widetilde{t}_i \in (0, \bar{t})$ for all $i = 2, \cdots, k^*$ such that

$$\boldsymbol{d}\big(h(\widetilde{\boldsymbol{r}}, \widetilde{\boldsymbol{\theta}}, \widetilde{\boldsymbol{t}}), \mathcal{G}\big) > 100 l^* \bar{\epsilon}$$

where $h$ is the mapping defined in eq. (J.38). Let $\widetilde{y} \triangleq h(\widetilde{\boldsymbol{r}}, \widetilde{\boldsymbol{\theta}}, \widetilde{\boldsymbol{t}})$. Moreover, the continuity of mapping $h$ implies the existence of some $\Delta > 0$ such that $(\widetilde{\boldsymbol{r}}, \widetilde{\boldsymbol{\theta}}, \widetilde{\boldsymbol{t}}) \in \mathcal{X} \subseteq h^{-1}\big(B(\widetilde{y}, \bar{\epsilon})\big)$ where the open domain $\mathcal{X}$ is defined as

$$\mathcal{X} \triangleq \{(\boldsymbol{r}, \boldsymbol{\theta}, \boldsymbol{t}) : t_1 = 0 \text{ and } \left\|\theta_i - \widetilde{\theta}_i\right\| < \Delta, |r_i - \widetilde{r}_i| < \Delta, |t_i - \widetilde{t}_i| < \Delta \,\forall i \in [k^*]\}.$$

In particular, the parameter $\Delta$ can be chosen small enough with

$$\Delta < \min_{i=2,\cdots,k^*} \widetilde{t}_i, \ \Delta < \min_{i=1,\cdots,k^*} \widetilde{r}_i \wedge 1$$

so that for any $(\boldsymbol{r}, \boldsymbol{\theta}, \boldsymbol{t}) \in \mathcal{X}$, we have $t_i > 0$ for all $i = 2, \cdots, k^*$ and $r_i > 0$ for all $i \in [k^*]$. Fix such $\Delta > 0$. Meanwhile, from the definition of the measure $\mu(\cdot)$ in eq. (J.40)eq. (J.48) and the fact that $\widetilde{\theta}_i \in F_{\boldsymbol{j}_i}$ where the closed set $F_{\boldsymbol{j}_i}$ is the support of the probability measure $S_{\boldsymbol{j}_i}$ on the unit sphere $\mathbb{S}^{d-1}$ (see Assumption I.6), one can see that $\mu(\mathcal{X}) > 0$.

Let $\boldsymbol{R} = (R_i)_{i=1}^{k^*}, \boldsymbol{\Theta} = (\Theta_i)_{i=1}^{k^*}, \boldsymbol{T} = (T_i)_{i=1}^{k^*}$ with $R_i = \eta \|W_i^{\eta}(\delta)\|$, $\Theta_i = W_i^{\eta}(\delta) / \|W_i^{\eta}(\delta)\|$ and $T_i = \eta(T_i^{\eta}(\delta) - T_{i-1}^{\eta}(\delta))$ for all $i \geq 2$ and $T_1 = 0$. Now recall Definition J.1 and consider the following event

$$B \triangleq \{(\boldsymbol{R}, \boldsymbol{\Theta}, \boldsymbol{T}) \in \mathcal{X}\} = \{Z_{T_1^{\eta}(\delta)} \text{ is of type-}(h(\mathcal{X}), \eta, \delta)\}.$$

Corollary J.14 then gives the bound

$$\liminf_{\eta \downarrow 0} \frac{\mathbb{P}(B)}{\widetilde{\lambda}(\eta)} \geq \delta^{\alpha_1} \mu(\mathcal{X}) > 0.$$

From now on, we fix some $c_* \in (0, \mu(\mathcal{X}))$. Given the fixed $\epsilon > 0$, we can define $\hat{t} = c_1 + c_1 \log(1/\epsilon)$ and choose $\widetilde{\epsilon} > 0$ as the largest possible value such that

$$\widetilde{\epsilon} \leq \frac{\epsilon}{4 \exp(2\hat{t}M)}, \ \bar{\rho}(\hat{t})\widetilde{\epsilon} \leq \epsilon, \ \widetilde{\rho}(\hat{t})\widetilde{\epsilon} \leq \epsilon$$

where functions $\bar{\rho}(\cdot), \widetilde{\rho}(\cdot)$ are defined in Corollary J.6 and J.9 respectively. We stress that $\bar{t}$ is a fixed constant while $\hat{t}$ depends on the value of $\epsilon$, and for any sufficiently small $\epsilon$ we will have $\hat{t} > \bar{t}$.

Let us define an event $A^{\times} = A_1^{\times} \cup A_2^{\times}$ where

$$A_1^{\times} \triangleq \left\{\exists n < T_1^{\eta}(\delta) \ s.t. \ \|X_n^{\eta}(x)\| > 3\epsilon\right\},$$

$$A_2^{\times} \triangleq \Big\{\exists j = 2, \cdots, k^* \ s.t.$$

$$\max_{k=1,2,\cdots,(T_j^{\eta}(\delta)-T_{j-1}^{\eta}(\delta)-1)\wedge(2\hat{t}/\eta)} \eta \left\|Z_{T_{j-1}^{\eta}(\delta)+1} + \cdots + Z_{T_{j-1}^{\eta}(\delta)+j}\right\| > \widetilde{\epsilon}\Big\}$$

where the positive integer $l^*$ is defined in eq. (J.1).

Now we focus on the event $B \backslash A^{\times}$. On one hand, Lemma J.15 shows that, for all $\eta$ sufficiently small, on this event we have $\|X_k^{\eta}\| < 3\epsilon < \bar{\epsilon}$ for all $k < T_1^{\eta}(\delta)$ and

$$\sup_{s \in [0, T_{k^*}^{\eta}(\delta)-T_1^{\eta}(\delta)]} \left\|X_{\lfloor s \rfloor + T_1^{\eta}(\delta}^{\eta} - \boldsymbol{z}^{\eta}(s)\right\| < 2\epsilon < \bar{\epsilon}.$$

Here

$$\boldsymbol{z}^{\eta}(s) = \widetilde{\boldsymbol{x}}^{\eta}\big(s, X^{(1)}; \boldsymbol{T}, \boldsymbol{W}\big) \,\forall s \geq 0$$

with $\boldsymbol{T} = \big(0, T_2^\eta(\delta) - T_1^\eta(\delta), T_3^\eta(\delta) - T_2^\eta(\delta), \cdots, T_{k^*}^\eta(\delta) - T_{k^*-1}^\eta(\delta)\big)$, $\boldsymbol{W} = \big(W_1^\eta(\delta), W_2^\eta(\delta), \cdots, W_{k^*}^\eta(\delta)\big)$, and $X^{(1)} = X_{T_1^\eta(\delta)-1}^\eta$. To better control the location of $\boldsymbol{z}^\eta(s)$, we also construct

$$\boldsymbol{z}_0^\eta(s) = \widetilde{\boldsymbol{x}}^\eta\big(s, \boldsymbol{0}; \boldsymbol{T}, \boldsymbol{W}\big) \ \forall s \geq 0$$

where the only difference is that we substitute the initial value $X^{(1)}$ with $\boldsymbol{0}$.

First of all, from eq. (J.20), we see that

$$\boldsymbol{d}(\boldsymbol{z}^\eta(s), \mathcal{G}^c) > 100 l^* \bar{\epsilon}, \ \boldsymbol{d}(\boldsymbol{z}_0^\eta(s), \mathcal{G}^c) > 100 l^* \bar{\epsilon} \ \forall s \in [0, \sum_{i=1}^{k^*} \boldsymbol{T}_i).$$

Besides, due to $\mathcal{X} \subseteq h^{-1}\big(B(\widetilde{y}, \bar{\epsilon})\big)$, we have that $\boldsymbol{d}\big(\boldsymbol{z}_0^\eta\big(\sum_{i=1}^{k^*} \boldsymbol{T}_i\big), \mathcal{G}\big) > 99 l^* \bar{\epsilon}$. Next, using Corollary J.7, we have that $\sup_{\in[0, \sum_{i=1}^{k^*} \boldsymbol{T}_i]} \|\boldsymbol{z}^\eta(s) - \boldsymbol{z}_0^\eta(s)\| < 3\rho^* \epsilon < \bar{\epsilon}$, which further implies that

$$\boldsymbol{d}\big(\boldsymbol{z}^\eta\big(\sum_{i=1}^{k^*} \boldsymbol{T}_i\big), \mathcal{G}\big) > 99 l^* \bar{\epsilon} - \bar{\epsilon} \geq 98 l^* \bar{\epsilon}.$$

Now we have the following facts regarding the distance between SGD iterates $X_j^\eta$ and the domain $\mathcal{G}$. First, on event $B \backslash A^\times$, it holds that

$$\boldsymbol{d}(X_j^\eta, \mathcal{G}^c) > 100 l^* \bar{\epsilon} - \bar{\epsilon} \geq 99 l^* \bar{\epsilon} \ \forall j < T_{k^*}^\eta(\delta),$$

implying that $\sigma(\eta) \geq T_{k^*}^\eta(\delta)$.

Next, if $R(\epsilon, \delta, \eta) < T_{k^*}^\eta(\delta)$, then there exists some $T_1^\eta(\delta \leq j < T_{k^*}^\eta(\delta)$ such that $\|X_j^\eta\| \leq 2\epsilon$ hence $\|\boldsymbol{z}^\eta(j - T_1^\eta)\| \leq 2\epsilon + 2\epsilon < \bar{\epsilon} < 100 l^* \bar{\epsilon}$. However, in light of eq. (J.20) we know that on event $\{R(\epsilon, \delta, \eta) < T_{k^*}^\eta(\delta)\} \cap (B \backslash A^\times)$, we have

$$\boldsymbol{d}\big(\boldsymbol{z}^\eta\big(\sum_{i=1}^{k^*} \boldsymbol{T}_i\big), \mathcal{G}^c\big) > 100 l^* \bar{\epsilon}$$

and yield a contradiction. Therefore, on event $B \backslash A^\times$ we must have $R(\epsilon, \delta, \eta) \geq T_{k^*}^\eta(\delta)$.

Besides, bounding the gap between $X_j^\eta, \boldsymbol{z}^\eta(j)$ and $\boldsymbol{z}_0^\eta(j)$ at time $j = T_{k^*}^\eta(\delta)$ using results above, we can show that

$$\boldsymbol{d}(X_{T_{k^*}^\eta(\delta)}^\eta, \mathcal{G}) > 98 l^* \bar{\epsilon} - \bar{\epsilon} \geq 97 l^* \bar{\epsilon}.$$

Therefore, we must have $\sigma(\eta) = T_{k^*}^\eta(\delta)$. In summary, we have shown that, for all $\eta$ sufficiently small, $B \backslash A^\times \subseteq A^\circ(\epsilon, \delta, \eta)$. Combining all the bounds above, we have

$$\liminf_{\eta \downarrow 0} \frac{\inf_{\|x\| \leq 2\epsilon} \mathbb{P}_x(A^\circ(\epsilon, \delta, \eta))}{\widetilde{\lambda}(\eta)} \geq \liminf_{\eta \downarrow 0} \frac{\mathbb{P}(B)}{\widetilde{\lambda}(\eta)} - \limsup_{\eta \downarrow 0} \frac{\sup_{\|x\| \leq 2\epsilon} \mathbb{P}(A^\times)}{\widetilde{\lambda}(\eta)}$$

$$\geq \delta^{\alpha_1} c_* - \limsup_{\eta \downarrow 0} \frac{\sup_{\|x\| \leq 2\epsilon} \mathbb{P}(A^\times)}{\widetilde{\widetilde{\lambda}}(\eta)}.$$

To conclude the proof, it suffices to invoke Lemma J.10 and J.12 with some $N > 1 + J_{\mathcal{G}} - \alpha_1$. $\square$

### J.4 PROOF OF THE MAIN RESULT

Now we are ready to state the main result and provide upper and lower bounds to the distribution of the scaled first exit time from domain $\mathcal{G}$. In particular, we define a scaling function

$$\lambda(\eta) \triangleq \widetilde{\lambda}(\eta) H(1/\eta) \tag{J.56}$$

where the function $\widetilde{\lambda}(\cdot)$ is defined in eq. (J.50). One can easily see that $\lambda \in RV_{1+J_{\mathcal{G}}}(\eta)$. We show that the scaled first exit time $\lambda(\eta)\sigma(\eta)$ converges in distribution to an exponential random variable as $\eta \downarrow 0$, which implies that, in expectation, the first exit time is roughly a $1/\eta^{1+J_{\mathcal{G}}}$ term and its order is dictated by the minimum cost for exit $J_{\mathcal{G}}$.

**Proposition J.23.** *Given any $C \in (0,1), u > 0$, the following inequalities hold for all $\epsilon > 0$ sufficiently small,*

$$\limsup_{\eta \downarrow 0} \sup_{\|x\| \leq 2\epsilon} \mathbb{P}_x(\sigma(\eta)\lambda(\eta) > u) \leq 2C + \exp(-(1-C)^2 qu),$$

$$\liminf_{\eta \downarrow 0} \inf_{\|x\| \leq 2\epsilon} \mathbb{P}_x(\sigma(\eta)\lambda(\eta) > u) \geq -C + \exp(-(1+C)qu)$$

*where $q = \mu\big(h^{-1}(\mathcal{G}^c)\big)$.*

Before presenting the proof to Proposition J.23, we make some preparations. First, we introduce stopping times (for all $k \geq 1$)

$$\tau_k(\epsilon, \delta, \eta) = \min\{n > \widetilde{\tau}_{k-1}(\epsilon, \delta, \eta) : \eta\|Z_n\| > \delta\}$$

$$\widetilde{\tau}_k(\epsilon, \delta, \eta) = \min\{n \geq \tau_k(\epsilon, \delta, \eta) : \|X_n^\eta\| \leq 2\epsilon\}$$

with the convention that $\tau_0(\epsilon, \delta, \eta) = \widetilde{\tau}_0(\epsilon, \delta, \eta) = 0$. Intuitively, at each $\widetilde{\tau}_k$ the SGD iterates have just returned to a small neighborhood of the local minimum $\mathbf{0}$. Due to Markov property of $X_n^\eta$, the SGD iterates almost *regenerate* at each $\widetilde{\tau}_k$ despite the previous trajectory, and the times $(\widetilde{\tau}_k)_{k\geq 1}$ partitions the entire timeline into different segments that can almost be interpreted as *regeneration cycles*, where $\widetilde{\tau}_{k-1}$ can be understood as the starting point of the $k-$th cycle. In light of the embedded (informal) regeneration process, one natural approach is to determine the dynamics and probability of making an exit on one (hence every) cycle, and this will be carried out with the help of technical results in the previous section. It is worth noticing that $\widetilde{\tau}_k$ are defined under the proviso that $\widetilde{\tau}_k \geq \tau_k$ where $\tau_k$ is the first big jump during the $k-$th cycle. Considering results such as Lemma J.12, it is reasonable to expect that the SGD iterates would be trapped at local minimum until a large jump strikes. Therefore, we define the (informal) regeneration points $\widetilde{\tau}_k$ in such a way that a cycle $[\widetilde{\tau}_{k-1}, \widetilde{\tau}_k)$ ends only if we have observed at least one large jump already. Regarding the notations, we add a remark that when there is no ambiguity we will drop the dependency on $\epsilon, \delta, \eta$ and simply write $\tau_k, \widetilde{\tau}_k$.

Specifically, we are interested in the large jumps and the accumulated cost thereof during each cycle, which definitely entails some systematic bookkeeping. For all $k \geq 1$, the random variable

$$\mathbf{j}_k \triangleq \#\{n = \tau_{k-1}(\epsilon, \delta, \eta), \tau_{k-1}(\epsilon, \delta, \eta) + 1, \cdots, \widetilde{\tau}_k(\epsilon, \delta, \eta) \wedge \sigma(\eta) : \eta\|Z_n\| > \delta\}$$

can be understood as the count of large jumps during the $k-$th cycle. Here are two remarks on this definition.

- First, for any $k$ with $\sigma(\eta) < \widetilde{\tau}_k$, we have $\mathbf{j}_k = 0$. Recall that the object we study here is the first exit time $\sigma(\eta)$, so there is virtually no need to keep track of the dynamics of the SGD after it leaves the domain $\mathcal{G}$.

- The random variable $\mathbf{j}_k$ is measurable w.r.t. $\mathcal{F}_{\widetilde{\tau}_k \wedge \sigma(\eta)}$, the stopped $\sigma-$algebra generated by the stopping time $\widetilde{\tau}_k \wedge \sigma(\eta)$, which, intuitively, is stating that one should be able to determine the number of large jumps during the $k-$th cycle when this cycle ends.

Furthermore, for each $k = 1, 2, \cdots$, let

$$T_{k,1}(\epsilon, \delta, \eta) \triangleq \tau_{k-1}(\epsilon, \delta, \eta) \wedge \sigma(\eta),$$

$$T_{k,j}(\epsilon, \delta, \eta) \triangleq \min\left\{n > T_{k,j-1}(\epsilon, \delta, \eta) : \eta\|Z_n\| > \delta\right\} \wedge \sigma(\eta) \wedge \widetilde{\tau}_k \ \forall j \geq 2,$$

$$W_{k,j}(\epsilon, \delta, \eta) \triangleq Z_{T_{k,j}(\epsilon, \delta, \eta)} \ \forall j \geq 1$$

with the convention $T_{k,0}(\epsilon, \delta, \eta) = \widetilde{\tau}_{k-1}(\epsilon, \delta, \eta)$. Note that for any $k \geq 1, j \geq 1, T_{k,j}$ is a stopping time. Besides, for any $k$ with $\widetilde{\tau}_k < \sigma(\eta)$

$$\widetilde{\tau}_{k-1} + 1 \leq T_{k,j} \leq \widetilde{\tau}_k \wedge \sigma(\eta) \ \forall j \in [\mathbf{j}_k], \tag{J.57}$$

and the sequences $(T_{k,j})_{j=1}^{\mathbf{j}_k}$ and $(W_{k,j})_{j=1}^{\mathbf{j}_k}$ are the arrival times and sizes of *large* jumps during the $k-$th cycle, respectively. Again, when there is no ambiguity we will drop the dependency on $\epsilon, \delta, \eta$ and simply write $T_{k,j}$ and $W_{k,j}$.

We are now able to keep track of the accumulated cost of jumps during each cycle. For any $k \geq 1, i \geq 1$, we define

$$\mathcal{J}_k^{\text{cycle}}(i; \epsilon, \delta, \eta) \triangleq \sum_{l=1}^i J(W_{k,i}(\epsilon, \delta, \eta)).$$

Moreover, we define the following index

$$\mathbf{j}^{\geq J_{\mathcal{G}}}(k; \epsilon, \delta, \eta) \triangleq \begin{cases} \min\{j = 1, 2, \cdots, \mathbf{j}_k : \mathcal{J}_k^{\text{cycle}}(j; \epsilon, \delta, \eta) \geq J_{\mathcal{G}}\} & \text{if } \widetilde{\tau}_k(\epsilon, \delta, \eta) < \sigma(\eta) \\ \infty & \text{otherwise} \end{cases}$$

with the convention that $\min \emptyset = \infty$. That is to say, if $\widetilde{\tau}_k \geq \sigma(\eta)$ or $\mathcal{J}_k^{\text{cycle}}(\mathbf{j}_k) < J_{\mathcal{G}}$, we define $\mathbf{j}^{\geq J_{\mathcal{G}}}(k)$ as $\infty$; otherwise $\mathbf{j}^{\geq J_{\mathcal{G}}}(k)$ is the index for the first large jump during the $k-$th cycle that drives the accumulated cost of large jumps to reach $J_{\mathcal{G}}$. For clarity of the presentation below, we also define

$$T^{\geq J_{\mathcal{G}}}(k; \epsilon, \delta, \eta) \triangleq \mathbb{1}\{\mathbf{j}^{\geq J_{\mathcal{G}}}(k) < \infty\} T_{k, \mathbf{j}^{\geq J_{\mathcal{G}}}(k)} + \mathbb{1}\{\mathbf{j}^{\geq J_{\mathcal{G}}}(k) = \infty\}0,$$

$$X^{\geq J_{\mathcal{G}}}(k; \epsilon, \delta, \eta) \triangleq \mathbb{1}\{\mathbf{j}^{\geq J_{\mathcal{G}}}(k) < \infty\} X^{\eta}_{T^{\geq J_{\mathcal{G}}}(k; \epsilon, \delta, \eta)} + \mathbb{1}\{\mathbf{j}^{\geq J_{\mathcal{G}}}(k) = \infty\}\mathbf{0}.$$

In other words, when $\mathbf{j}^{\geq J_{\mathcal{G}}}(k) < \infty$, the random variable $X^{\geq J_{\mathcal{G}}}(k; \epsilon, \delta\eta)$ is equal to $X^{\eta}_{T_{k, \mathbf{j}^{\geq J_{\mathcal{G}}}(k)}}$, the location of the SGD iterates right when the accumulated cost on the $k-$th cycle reaches $J_{\mathcal{G}}$. Notation wise, we will drop the dependency on $\epsilon, \delta, \eta$ again and simply write $\mathbf{j}^{\geq J_{\mathcal{G}}}(k), X^{\geq J_{\mathcal{G}}}(k)$ or $\mathcal{J}_k^{\text{cycle}}(j)$ when there is no risk of ambiguity.

As have mentioned above, we want to zoom in on the each cycle and analyze the probability of each possible case. To be specific, we want to introduce a series of scenarios, formally defined as several events, that exhaust all the possibilities during a cycle. The events will be denoted in the form of $\mathbf{A}_k^{\circ}$ or $\mathbf{B}_j^{\times}$ largely following the next few rules. First, we say an event/scenario is *atypical* if its probability is rather small, and we assign it with superscript $\times$; otherwise we say the event is *typical* and add a superscript $\circ$. Besides, the subscript $k$ indicates the cycle that the event concerns. Lastly, events with label (of type) $\mathbf{A}$ usually describe how the SGD iterates try to escape from $\mathcal{G}$, whereas events with label $\mathbf{B}$ focuses on how the SGD iterates return to the local minimum.

Now we proceed and formally define the said series of events. First, for each $k \geq 1$, define the event

$$\mathbf{A}_{k,0}^{\times}(\epsilon, \delta, \eta) \triangleq \left\{\exists i \in [l^*] \text{ s.t.} \max_{j = T_{k,i-1}+1, T_{k,i-1}+2, \cdots, (T_{k,i}-1) \wedge (T_{k,i-1}+\lceil\frac{2\hat{t}(\epsilon)}{\eta}\rceil) \wedge \widetilde{\tau}_k \wedge \sigma(\eta)} \eta \left\|Z_{T_{k,i}+1} + \cdots + Z_j\right\| > \widetilde{\epsilon}(\epsilon)\right\}$$

$$\cup \left\{\left\|X_j^{\eta}\right\| > 3\epsilon \text{ for some } j = \widetilde{\tau}_{k-1}, \widetilde{\tau}_{k-1} + 1, \cdots, T_{k,1} - 1\right\} \tag{J.58}$$

with $\hat{t}(\epsilon) = c_1 + c_1 \log(1/\epsilon)$ and function $\widetilde{\epsilon}(\cdot)$ defined in eq. (J.51). It is similar to the event $A^{\times}$ defined in eq. (J.52)eq. (J.53) in the previous section, and its probability will be controlled using a similar approach.

Next, consider event (for all $k \geq 1$)

$$\mathbf{A}_{k,1}^{\times}(\epsilon, \delta, \eta) \triangleq \left\{\sigma(\eta) < \widetilde{\tau}_k, \ \mathcal{J}_k^{\text{cycle}}(\mathbf{j}_k) < J_{\mathcal{G}}\right\} \tag{J.59}$$

that describes an atypical case where the exit occurs during the $k-$th cycle yet the accumulated cost of all large jumps is less than $J_{\mathcal{G}}$. Another atypical event is defined as

$$\mathbf{A}_{k,2}^{\times} \triangleq \{\mathbf{j}^{\geq J_{\mathcal{G}}}(k) < \infty\} \cap \{\exists j = 2, 3, \cdots, \mathbf{j}^{\geq J_{\mathcal{G}}}(k) \text{ s.t. } T_{k,j} - T_{k,j-1} > 2\hat{t}(\epsilon)/\eta\}. \tag{J.60}$$

This event describes the case where the accumulated cost of large jumps in a cycle has reached $J_{\mathcal{G}}$ but the inter-arrival time between some large jumps are unusually long. We stress that, by definition of $\mathbf{j}^{\geq J_{\mathcal{G}}}(k)$, we have $\{\mathbf{j}^{\geq J_{\mathcal{G}}}(k) < \infty\} = \{\widetilde{\tau}_{k-1} < \sigma(\eta)\} \cap \{\mathcal{J}_k^{\text{cycle}}(\mathbf{j}_k) \geq J_{\mathcal{G}}\}$.

Moving on, we consider the following events (defined for all $k \geq 1$)

$$\mathbf{A}_{k,3}^{\times} \triangleq \{\mathcal{J}_k^{\text{cycle}}(\mathbf{j}_k) < J_{\mathcal{G}}, \ \widetilde{\tau}_k < \sigma(\eta)\} \cap \left\{\exists j = 2, 3, \cdots, \mathbf{j}_k \text{ s.t. } T_{k,j} - T_{k,j-1} > 2\hat{t}(\epsilon)/\eta\right\} \tag{J.61}$$

that describes another similar case where the $k-$th cycle ends with return to the local minimum but the inter-arrival time between some large jumps are unusually long.

The next event

$$\mathbf{A}_{k,4}^{\times} \triangleq \{\mathbf{j}^{\geq J_{\mathcal{G}}}(k) < \infty, \ \mathcal{J}_k^{\text{cycle}}(\mathbf{j}^{\geq J_{\mathcal{G}}}(k)) = J_{\mathcal{G}}\} \cap \{d(X^{\geq J_{\mathcal{G}}}(k), \mathcal{G}^c) \leq \bar{\epsilon}\} \cap \left\{\exists j \in [\mathbf{j}^{\geq J_{\mathcal{G}}}(k)] \text{ s.t. } \eta \|W_{k,j}\| \leq \bar{\delta}\right\} \tag{J.62}$$

describes the case where the accumulated cost on the $k-$th cycle has hit $J_{\mathcal{G}}$ exactly at some point with SGD iterates being rather close to $\mathcal{G}^c$ yet some large jumps are not large enough when compared to the fixed constant $\bar{\delta}$.

Meanwhile, with events

$$\mathbf{A}_{k,5}^{\times} \triangleq \{\mathbf{j}^{\geq J_{\mathcal{G}}}(k) < \infty, \ \mathcal{J}_k^{\text{cycle}}(\mathbf{j}^{\geq J_{\mathcal{G}}}(k)) = J_{\mathcal{G}}\} \cap \left\{\boldsymbol{d}(X^{\geq J_{\mathcal{G}}}(k), \partial\mathcal{G}) < \epsilon\right\}, \tag{J.63}$$

we analyze the case where the SGD iterates reaches somewhere close the the boundary set $\partial\mathcal{G}$ with large jumps of cost equal to $J_{\mathcal{G}}$. Lastly, define event

$$\mathbf{A}_{k,6}^{\times} \triangleq \{\mathbf{j}^{\geq J_{\mathcal{G}}}(k) < \infty, \ \mathcal{J}_k^{\text{cycle}}(\mathbf{j}^{\geq J_{\mathcal{G}}}(k)) > J_{\mathcal{G}}\} \tag{J.64}$$

for the case where the accumulated cost of large jumps on the $k-$th cycle exceeds $J_{\mathcal{G}}$ without hitting it. As an amalgamation of these atypical scenarios, we let

$$\mathbf{A}_k^{\times}(\epsilon, \delta, \eta) \triangleq \bigcup_{i=0}^{6} \mathbf{A}_{k,i}^{\times}(\epsilon, \delta, \eta). \tag{J.65}$$

Next, we analyze the probability of some events $(\mathbf{B}_k^{\times})_{k \geq 1}$ that concern the behavior of SGD iterates during the $k-$th cycle after $T^{\geq J_{\mathcal{G}}}(k)$. Let us define

$$\mathbf{B}_{k,1}^{\times}(\epsilon, \delta, \eta) \triangleq \{\mathbf{j}^{\geq J_{\mathcal{G}}}(k) < \infty, \ \mathcal{J}_k^{\text{cycle}}(\mathbf{j}^{\geq J_{\mathcal{G}}}(k)) = J_{\mathcal{G}}\} \cap \{\boldsymbol{d}(X^{\geq J_{\mathcal{G}}}(k), \mathcal{G}^c) \geq \epsilon\}$$

$$\cap \left\{T_{k,j} - T_{k,j-1} \leq 2\frac{\hat{t}(\epsilon)}{\eta} \ \forall j = 2, 3, \cdots, \mathbf{j}^{\geq J_{\mathcal{G}}}(k)\right\}$$

$$\mathbf{B}_{k,2}^{\times}(\epsilon, \delta, \eta) \triangleq \{\widetilde{\tau}_k - T_{k,\mathbf{j}^{\geq J_{\mathcal{G}}}(k)} > \rho(\epsilon)/\eta\} \cup \{\sigma(\eta) < \widetilde{\tau}_k\}$$

$$\mathbf{B}_k^{\times}(\epsilon, \delta, \eta) \triangleq \mathbf{B}_{k,1}^{\times} \cap \mathbf{B}_{k,2}^{\times} \tag{J.66}$$

where $\rho(\cdot)$ is the function in Lemma J.11. From the definition of $\mathbf{B}_k^{\times}$, in particular the inclusion of $\mathbf{B}_{k,2}^{\times}$, one can see that the intuitive interpretation of event $\mathbf{B}_k^{\times}$ is that the SGD iterates *did not return* to local minimum efficiently or even escaped from $\mathcal{G}$ after large jumps with cost equal to $J_{\mathcal{G}}$. In comparison, the following events will characterize what would typically happen during each attempt:

$$\mathbf{A}_k^{\circ}(\epsilon, \delta, \eta) \triangleq \{\mathbf{j}^{\geq J_{\mathcal{G}}}(k) < \infty, \ \mathcal{J}_k^{\text{cycle}}(\mathbf{j}^{\geq J_{\mathcal{G}}}(k)) = J_{\mathcal{G}}\} \cap \{\sigma(\eta) = T^{\geq J_{\mathcal{G}}}(k)\}$$

$$\cap \left\{T_{k,j} - T_{k,j-1} \leq \frac{2\hat{t}(\epsilon)}{\eta} \ \forall j = 2, 3, \cdots, \mathbf{j}^{\geq J_{\mathcal{G}}}(k)\right\}, \tag{J.67}$$

$$\mathbf{B}_k^{\circ}(\epsilon, \delta, \eta) \triangleq \{\sigma(\eta) > \widetilde{\tau}_k, \ \widetilde{\tau}_k - T_{k,1} \leq \frac{2l^*\hat{t}(\epsilon) + \rho(\epsilon)}{\eta}\} \cap \{\mathcal{J}_k^{\text{cycle}}(\mathbf{j}_k) \leq J_{\mathcal{G}}\}. \tag{J.68}$$

Intuitively speaking, $\mathbf{A}_k^{\circ}$ tells us that the exit happened right at $T^{\geq J_{\mathcal{G}}}(k)$ with accumulated cost of large jumps in the cycle being exactly $J_{\mathcal{G}}$, and $\mathbf{B}_k^{\circ}$ requires that that the first exit did not occur during the $k-$th cycle, and the SGD iterates returned to local minimum rather efficiently. It is worth noticing that, by definition, $\mathbf{j}^{\geq J_{\mathcal{G}}}(k) < l^*$ if $\mathbf{j}^{\geq J_{\mathcal{G}}}(k) < \infty$.

Our immediate next goal is to analyze the event

$$\mathbf{A}^{\times}(\epsilon, \delta, \eta) \triangleq \bigcup_{k \geq 1} \left(\bigcap_{i=1}^{k-1} \left(\mathbf{A}_i^{\times} \cup \mathbf{B}_i^{\times} \cup \mathbf{A}_i^{\circ}\right)^c\right) \cap \left(\mathbf{A}_k^{\times} \cup \mathbf{B}_k^{\times}\right). \tag{J.69}$$

In particular, we show that its probability can be made arbitrarily small with proper $\epsilon, \delta, \eta$, implying that we will almost always observe the event $(\mathbf{A}^{\times})^c$.

**Lemma J.24.** *Given any $C > 0$, the following claim holds for all $\epsilon > 0, \delta > 0$ sufficiently small:*

$$\limsup_{\eta \downarrow 0} \ \sup_{x: \|x\| \leq 2\epsilon} \ \mathbb{P}_x(\mathbf{A}^{\times}(\epsilon, \delta, \eta)) \leq C.$$

*Proof.* We start by bounding for the probabilities of all the $\mathbf{A}_{k,i}^{\times}$ events. Fix some $N > 1 + J_{\mathcal{G}}$. For

$$\mathbf{A}_{k,0}^{\times}(\epsilon, \delta, \eta) = \left\{\exists i \in [l^*] \ s.t. \max_{j = T_{k,i-1}+1, T_{k,i-1}+2, \cdots, (T_{k,i}-1) \wedge (T_{k,i-1}+\lceil\frac{2\hat{t}(\epsilon)}{\eta}\rceil) \wedge \widetilde{\tau}_k \wedge \sigma(\eta)} \eta \left\|Z_{T_{k,i}+1} + \cdots + Z_j\right\| > \widetilde{\epsilon}(\epsilon)\right\}$$

$$\cup \left\{\left\|X_j^{\eta}\right\| > 3\epsilon \ \text{for some} \ j = \widetilde{\tau}_{k-1}, \widetilde{\tau}_{k-1} + 1, \cdots, T_{k,1} - 1\right\},$$

using Lemma J.10 and J.12, one can show the existence of some $\epsilon_0 \in (0, \bar{\epsilon}/3)$ such that for any $\epsilon \in (0, \epsilon_0)$, there is some $\delta_0(\epsilon) > 0$ with

$$\limsup_{\eta} \frac{\sup_{\|x\| \leq 2\epsilon} \mathbb{P}_x(\mathbf{A}_{1,0}^{\times}(\epsilon, \delta, \eta))}{\eta^N} = 0 \quad \forall \delta \in (0, \delta_0(\epsilon)). \tag{J.70}$$

Next, for $\mathbf{A}_{k,1}^{\times}(\epsilon, \delta, \eta) = \left\{ \sigma(\eta) < \widetilde{\tau}_k, \ \mathcal{J}_k^{\text{cycle}}(\mathbf{j}_k) < J_{\mathcal{G}} \right\}$, it follows from Lemma J.17 that, for any $\epsilon \in (0, \epsilon_0)$, there exists some $0 < \delta_1(\epsilon) \leq \delta_0(\epsilon)$ such that

$$\limsup_{\eta} \frac{\sup_{\|x\| \leq 2\epsilon} \mathbb{P}_x(\mathbf{A}_{1,1}^{\times}(\epsilon, \delta, \eta))}{\eta^N} = 0 \quad \forall \delta \in (0, \delta_1(\epsilon)). \tag{J.71}$$

For event $\mathbf{A}_{k,2}^{\times} = \{ \boldsymbol{j}^{\geq J_{\mathcal{G}}}(k) < \infty \} \cap \{ \exists j = 2, 3, \cdots, \boldsymbol{j}^{\geq J_{\mathcal{G}}}(k) \ s.t. \ T_{k,j} - T_{k,j-1} > 2\hat{t}(\epsilon)/\eta \}$, Lemma J.18 implies that, for any $\epsilon \in (0, \epsilon_0)$, there exists some $0 < \delta_2(\epsilon) \leq \delta_1(\epsilon)$ such that

$$\limsup_{\eta} \frac{\sup_{\|x\| \leq 2\epsilon} \mathbb{P}_x(\mathbf{A}_{1,2}^{\times}(\epsilon, \delta, \eta))}{\eta^N} = 0 \quad \forall \delta \in (0, \delta_2(\epsilon)). \tag{J.72}$$

For event $\mathbf{A}_{k,3}^{\times} = \{ \mathcal{J}_k^{\text{cycle}}(\mathbf{j}_k) < J_{\mathcal{G}}, \ \widetilde{\tau}_k < \sigma(\eta) \} \cap \left\{ \exists j = 2, 3, \cdots, \mathbf{j}_k \ s.t. \ T_{k,j} - T_{k,j-1} > 2\hat{t}(\epsilon)/\eta \right\}$, thanks to Lemma J.19, one can see that, for any $\epsilon \in (0, \epsilon_0)$, there exists some $0 < \delta_3(\epsilon) \leq \delta_2(\epsilon)$ such that

$$\limsup_{\eta} \frac{\sup_{\|x\| \leq 2\epsilon} \mathbb{P}_x(\mathbf{A}_{1,3}^{\times}(\epsilon, \delta, \eta))}{\eta^N} = 0 \quad \forall \delta \in (0, \delta_3(\epsilon)). \tag{J.73}$$

For event

$$\mathbf{A}_{k,4}^{\times} = \{ \mathbf{j}^{\geq J_{\mathcal{G}}}(k) < \infty, \ \mathcal{J}_k^{\text{cycle}}(\mathbf{j}^{\geq J_{\mathcal{G}}}(k)) = J_{\mathcal{G}} \} \cap \{ \boldsymbol{d}(X^{\geq J_{\mathcal{G}}}(k), \mathcal{G}^c) \leq \bar{\epsilon} \} \cap \left\{ \exists j \in [\boldsymbol{j}^{\geq J_{\mathcal{G}}}(k)] \ s.t. \ \eta \, \|W_{k,j}\| \leq \bar{\delta} \right\},$$

using Lemma J.20, one can see the existence of some $\Delta > 0$ such that, for all $\epsilon \in (0, \epsilon_0)$, there is some $0 < \delta_4(\epsilon) \leq \delta_3(\epsilon)$ such that

$$\limsup_{\eta} \frac{\sup_{\|x\| \leq 2\epsilon} \mathbb{P}_x(\mathbf{A}_{1,4}^{\times}(\epsilon, \delta, \eta))}{\eta^{1+J_{\mathcal{G}}-\alpha_1+\Delta}} = 0 \quad \forall \delta \in (0, \delta_4(\epsilon)). \tag{J.74}$$

As for event $\mathbf{A}_{k,5}^{\times} = \{ \mathbf{j}^{\geq J_{\mathcal{G}}}(k) < \infty, \ \mathcal{J}_k^{\text{cycle}}(\mathbf{j}^{\geq J_{\mathcal{G}}}(k)) = J_{\mathcal{G}} \} \cap \left\{ \boldsymbol{d}(X^{\geq J_{\mathcal{G}}}(k), \partial \mathcal{G}) < \epsilon \right\}$, the plan is to use Lemma J.21 to control its probability. But before that, we recall the definition of function $\Psi(\epsilon) = \mu \left( h^{-1} \left( (\partial \mathcal{G})^{(3+3\rho^*)\epsilon} \right) \right)$ in Lemma J.21 and make several observations. First, note that $\partial \mathcal{G}$ is a closed set, so $\cap_{\epsilon>0} h^{-1} \left( (\partial \mathcal{G})^{\epsilon} \right) = h^{-1}(\partial \mathcal{G})$ for the mapping $h$ in eq. (J.38). Then it follows from Assumption I.9 that $\lim_{\epsilon \downarrow 0} \mu \left( h^{-1} \left( (\partial \mathcal{G})^{\epsilon} \right) \right) = 0$, implying that for $\epsilon > 0$ sufficiently small, we must have

$$\Psi(\epsilon) < \frac{Cc_*}{4}.$$

where $C > 0$ is the fixed constant in the description of the Lemma, and $c_* > 0$ is the constant in Lemma J.22. We stress that both of them would not vary with $\epsilon, \delta, \eta$. Combining this with Lemma J.21, we know the existence of some $\epsilon_5 \in (0, \epsilon_0]$ such that, for all $\epsilon \in (0, \epsilon_5)$, one can find some $0 < \delta_5(\epsilon) \leq \delta_4(\epsilon)$ with

$$\limsup_{\eta} \frac{\sup_{\|x\| \leq 2\epsilon} \mathbb{P}_x(\mathbf{A}_{1,5}^{\times}(\epsilon, \delta, \eta))}{\widetilde{\lambda}(\eta)} \leq \delta^{\alpha} \Psi(\epsilon) < \delta^{\alpha} \frac{Cc_*}{4} \quad \forall \delta \in (0, \delta_5(\epsilon)). \tag{J.75}$$

Lastly, for event $\mathbf{A}_{k,6}^{\times} = \{ \mathbf{j}^{\geq J_{\mathcal{G}}}(k) < \infty, \ \mathcal{J}_k^{\text{cycle}}(\mathbf{j}^{\geq J_{\mathcal{G}}}(k)) > J_{\mathcal{G}} \}$, from Lemma J.20 again we know that (let $\epsilon_6 = \epsilon_5$) for all $\epsilon \in (0, \epsilon_6)$, one can find some $0 < \delta_6(\epsilon) \leq \delta_5(\epsilon)$ with

$$\limsup_{\eta} \frac{\sup_{\|x\| \leq 2\epsilon} \mathbb{P}_x(\mathbf{A}_{1,6}^{\times}(\epsilon, \delta, \eta))}{\eta^{1+J_{\mathcal{G}}-\alpha_1+\Delta}} = 0 \quad \forall \delta \in (0, \delta_6(\epsilon)). \tag{J.76}$$

Recall that $\mathbf{A}_k^\times(\epsilon, \delta, \eta) = \bigcup_{i=0}^6 \mathbf{A}_{k,i}^\times(\epsilon, \delta, \eta)$. Also, the events $\mathbf{B}_k^\times, \mathbf{A}_k^\circ, \mathbf{B}_k^\circ$ are defined in eq. (J.66)eq. (J.67)eq. (J.68) respectively. Before bounding the conditional probability of events $\mathbf{B}_k^\times$, we make several observations. First, if we consider the event $\bigcap_{j=1}^k (\mathbf{A}_j^\times \cup \mathbf{B}_j^\times)^c \cap \mathbf{B}_j^\circ$, the inclusion of the $(\mathbf{B}_j^\circ)_{j=1}^k$ implies that during the first $k$ cycles the SGD iterates have never left $\mathcal{G}$, so

$$\bigcap_{j=1}^k (\mathbf{A}_j^\times \cup \mathbf{B}_j^\times)^c \cap \mathbf{B}_j^\circ = \big( \bigcap_{j=1}^k (\mathbf{A}_j^\times \cup \mathbf{B}_j^\times)^c \cap \mathbf{B}_j^\circ \big) \cap \{\sigma(\eta) > \widetilde{\tau}_k\}.$$

Next, note that

$$\mathbb{P}_x\big(\mathbf{B}_k^\times \mid \bigcap_{j=1}^{k-1} (\mathbf{A}_j^\times \cup \mathbf{B}_j^\times)^c \cap \mathbf{B}_j^\circ\big)$$

$$= \underbrace{\mathbb{P}_x\big(\mathbf{B}_{k,1}^\times \mid \bigcap_{j=1}^{k-1} (\mathbf{A}_j^\times \cup \mathbf{B}_j^\times)^c \cap \mathbf{B}_j^\circ\big)}_{\triangleq (\mathrm{I})} \cdot \underbrace{\mathbb{P}_x\Big(\mathbf{B}_{k,2}^\times \mid \big( \bigcap_{j=1}^{k-1} (\mathbf{A}_j^\times \cup \mathbf{B}_j^\times)^c \cap \mathbf{B}_j^\circ \big) \cap \mathbf{B}_{k,1}^\times\Big)}_{\triangleq (\mathrm{II})}.$$

We start by analyzing term (I). From Assumption I.7, one can see that on event $\mathbf{B}_{k,1}^\times$, it holds that

$$\mathbf{j}^{\geq J_\mathcal{G}}(k) = k^*, \quad (W_{k,1}, \cdots, W_{k,k^*}) \text{ is of type-}\mathbf{j} \text{ for some } \mathbf{j} \in \mathbf{j}(i^*)$$

and $T_{k,j} - T_{k,j-1} \leq 2\frac{\hat{t}(\epsilon)}{\eta} \; \forall j = 2, 3, \cdots, k^*$. Due to independence of $W_{k,i}$ and $T_{k,i+1} - T_{k,i}$, we have

$$\sup_{\|x\| \leq 2} (\mathrm{I}) \leq \big(\mathbb{P}(T_1^\eta(\delta) \leq \frac{2\hat{t}(\epsilon)}{\eta})\big)^{k^*-1} \cdot \sum_{\mathbf{j}: \; \mathbf{j} \in \mathbf{j}(i^*)} \mathbb{P}\Big((W_1^\eta(\delta), \cdots, W_{k^*}^\eta(\delta)) \text{ is of type-}\mathbf{j}\Big)$$

$$= \big(\mathbb{P}(T_1^\eta(\delta) \leq \frac{2\hat{t}(\epsilon)}{\eta})\big)^{k^*-1} \cdot \sum_{\mathbf{j}: \; \mathbf{j} \in \mathbf{j}(i^*)} \prod_{i=1}^{k^*} \frac{H_{\mathbf{j}_i}(\delta/\eta)}{H(\delta/\eta)}.$$

Due to the strong Markov property at stopping time $\widetilde{\tau}_{k-1}$, Lemma G.4, and the regularly varying nature of functions $H_i, H$,

$$\limsup_{\eta \downarrow 0} \frac{\sup_{\|x\| \leq 2} (\mathrm{I})}{\big(2\hat{t}(\epsilon)/\delta^{\alpha_1}\big)^{k^*-1} \big(H(1/\eta)/\eta\big)^{k^*-1} \cdot |\mathbf{j}(i^*)| \cdot \frac{\delta^{k^* \alpha_1}}{\delta^{k^* + J_\mathcal{G}}} \cdot \big( \prod_{q=1}^m \big(H_i(1/\eta)\big)^{i_q^*} \big) / \big(H(1/\eta)\big)^{k^*}} \leq 1.$$

In other words, for any $\epsilon, \delta > 0$, there exists some $C(\epsilon, \delta) < \infty$ such that

$$\limsup_{\eta \downarrow 0} \frac{\sup_{\|x\| \leq 2} (\mathrm{I})}{\delta^{\alpha_1} \widetilde{\lambda}(\eta)} \leq |\mathbf{j}(i^*)| C(\epsilon, \delta).$$

On the other hand, due to Lemma J.11 and the strong Markov property at stopping time $T_{k,k^*}$, for any $\epsilon \in (0, \bar{\epsilon})$ and any $\delta > 0$,

$$\limsup_{\eta \downarrow 0} \sup_{\|x\| \leq 2\epsilon} (\mathrm{II}) < \frac{Cc_*/4}{|\mathbf{j}(i^*)|C(\epsilon, \delta)}.$$

In summary, we have established that for any $\epsilon \in (0, \bar{\epsilon})$ and any $\delta > 0$,

$$\limsup_{\eta \downarrow 0} \frac{\sup_{\|x\| \leq 2\epsilon} \mathbb{P}_x\big(\mathbf{B}_k^\times \mid \bigcap_{j=1}^{k-1} (\mathbf{A}_j^\times \cup \mathbf{B}_j^\times)^c \cap \mathbf{B}_j^\circ\big)}{\delta^{\alpha_1} \cdot \widetilde{\lambda}(\eta)} < Cc_*/4 \; \forall k \geq 1. \tag{J.77}$$

Similarly, we can bound conditional probabilities of the form $\mathbb{P}_x\big(\mathbf{A}_k^\times \mid \bigcap_{j=1}^{k-1} (\mathbf{A}_j^\times \cup \mathbf{B}_j^\times)^c \cap \mathbf{B}_j^\circ\big)$. To be specific, recall that $\mathbf{A}_k^\times = \cup_{i=0}^6 \mathbf{A}_{k,i}^\times$. Combining eq. (J.70)-eq. (J.76) with strong Markov property, we know the for all $\epsilon \in (0, \epsilon_6)$ and $k \geq 1$,

$$\limsup_{\eta \downarrow 0} \frac{\sup_{\|x\| \leq 2\epsilon} \mathbb{P}_x\big(\mathbf{A}_k^\times \mid \bigcap_{j=1}^{k-1} (\mathbf{A}_j^\times \cup \mathbf{B}_j^\times)^c \cap \mathbf{B}_j^\circ\big)}{\delta^{\alpha_1} \cdot \widetilde{\lambda}(\eta)} < Cc_*/4 \; \forall \delta \in (0, \delta_6(\epsilon)). \tag{J.78}$$

On the other hand, a lower bound can be established for conditional probability of event $\mathbf{A}_k^\circ$, the event defined in eq. (J.67). Due to Lemma J.22 and the strong Markov property at $\widetilde{\tau}_{k-1}$, one can see the existence of some $\epsilon_7 \in (0, \epsilon_6]$ such that, for any $\epsilon \in (0, \epsilon_7)$, there is some $0 < \delta_7(\epsilon) \leq \delta_6(\epsilon)$ such that

$$\liminf_{\eta \downarrow 0} \frac{\inf_{\|x\| \leq 2\epsilon} \mathbb{P}_x\big(\mathbf{A}_k^\circ \mid \bigcap_{j=1}^{k-1}(\mathbf{A}_j^\times \cup \mathbf{B}_j^\times)^c \cap \mathbf{B}_j^\circ\big)}{\delta^{\alpha_1}\widetilde{\lambda}(\eta)} \geq c_* \quad \forall \delta \in (0, \delta_7(\epsilon)), \ k \geq 1. \qquad \text{(J.79)}$$

In order to apply the bounds eq. (J.77)-eq. (J.79), we make use of the following inclusion relationship:

$$\big(\bigcap_{j=1}^{k-1}(\mathbf{A}_j^\times \cup \mathbf{B}_j^\times)^c \cap \mathbf{B}_j^\circ\big) \cap (\mathbf{A}_k^\times \cup \mathbf{B}_k^\times)^c \subseteq \mathbf{A}_k^\circ \cup \mathbf{B}_k^\circ. \qquad \text{(J.80)}$$

To see why this is true, let us consider a decomposition of the event on the L.H.S. of eq. (J.80). As mentioned above, on event $\bigcap_{j=1}^{k-1}(\mathbf{A}_j^\times \cup \mathbf{B}_j^\times)^c \cap \mathbf{B}_j^\circ$ we know that $\sigma(\eta) > \widetilde{\tau}_{k-1}$, implying that for the first $k-1$ cycles the first exit does not occur and leaving us with only three possibilities on this event:

- $\mathbf{j}^{\geq J_\mathcal{G}}(k) = \infty$ (on $\{\sigma(\eta) > \widetilde{\tau}_{k-1}\}$, this would happen if and only if $\mathcal{J}_k^{\text{cycle}}(\mathbf{j}_k) < J_\mathcal{G}$);

- $\mathbf{j}^{\geq J_\mathcal{G}}(k) < \infty, X^{\geq J_\mathcal{G}}(k) \notin \mathcal{G}$;

- $\mathbf{j}^{\geq J_\mathcal{G}}(k) < \infty, X^{\geq J_\mathcal{G}}(k) \in \mathcal{G}$.

Let us partition the said event accordingly and analyze them one by one.

- On $\big(\bigcap_{j=1}^{k-1}(\mathbf{A}_j^\times \cup \mathbf{B}_j^\times)^c \cap \mathbf{B}_j^\circ\big) \cap (\mathbf{A}_k^\times \cup \mathbf{B}_k^\times)^c \cap \{\mathcal{J}_k^{\text{cycle}}(\mathbf{j}_k) < J_\mathcal{G}\}$, due to the exclusion of $\mathbf{A}_k^\times$ (especially $\mathbf{A}_{k,1}^\times$ and $\mathbf{A}_{k,3}^\times$), we can see that if $\mathbf{j}_k < l^*$, then we must have $\sigma(\eta) > \widetilde{\tau}_k$ and $\widetilde{\tau}_k - T_{k,1} \leq 2l^*\hat{t}(\epsilon)/\eta$. In particular, note that on this event we must have $l^* > \mathbf{j}_k$ by definition of $l^*$ in eq. (J.1). Therefore,

$$\big(\bigcap_{j=1}^{k-1}(\mathbf{A}_j^\times \cup \mathbf{B}_j^\times)^c \cap \mathbf{B}_j^\circ\big) \cap (\mathbf{A}_k^\times \cup \mathbf{B}_k^\times)^c \cap \{\mathbf{j}^{\geq J_\mathcal{G}}(k) = \infty\} \subseteq \mathbf{B}_k^\circ.$$

- On $\big(\bigcap_{j=1}^{k-1}(\mathbf{A}_j^\times \cup \mathbf{B}_j^\times)^c \cap \mathbf{B}_j^\circ\big) \cap (\mathbf{A}_k^\times \cup \mathbf{B}_k^\times)^c \cap \{\mathbf{j}^{\geq J_\mathcal{G}}(k) < \infty, X^{\geq J_\mathcal{G}}(k) \notin \mathcal{G}\}$, the exclusion of $\mathbf{A}_{k,2}^\times$ implies that $T_{k,j} - T_{k,j-1} \leq 2\hat{t}(\epsilon)/\eta$ for all $j = 2, \cdots, \mathbf{j}^{\geq J_\mathcal{G}}(k)$, and the exclusion of $\mathbf{A}_{k,6}^\times$ tells us that $\mathcal{J}_k^{\text{cycle}}(\mathbf{j}^{\geq J_\mathcal{G}}(k)) = J_\mathcal{G}$. Moreover, the exclusion of $\mathbf{A}_{k,5}^\times \cup \mathbf{A}_{k,6}^\times$ tells us that if $X^{\geq J_\mathcal{G}}(k) \notin \mathcal{G}$, we must have $d(X^{\geq J_\mathcal{G}}(k), \mathcal{G}) > \epsilon$. In summary,

$$\big(\bigcap_{j=1}^{k-1}(\mathbf{A}_j^\times \cup \mathbf{B}_j^\times)^c \cap \mathbf{B}_j^\circ\big) \cap (\mathbf{A}_k^\times \cup \mathbf{B}_k^\times)^c \cap \{\mathbf{j}^{\geq J_\mathcal{G}}(k) < \infty, X^{\geq J_\mathcal{G}}(k) \notin \mathcal{G}\} \subseteq \mathbf{A}_k^\circ.$$

- On $\big(\bigcap_{j=1}^{k-1}(\mathbf{A}_j^\times \cup \mathbf{B}_j^\times)^c \cap \mathbf{B}_j^\circ\big) \cap (\mathbf{A}_k^\times \cup \mathbf{B}_k^\times)^c \cap \{\mathbf{j}^{\geq J_\mathcal{G}}(k) < \infty, X^{\geq J_\mathcal{G}}(k) \in \mathcal{G}\}$, the same argument in the previous bullet point can be applied to show that $T_{k,j} - T_{k,j-1} \leq 2\hat{t}(\epsilon)/\eta$ for all $j = 2, \cdots, \mathbf{j}^{\geq J_\mathcal{G}}(k)$, $\mathcal{J}_k^{\text{cycle}}(\mathbf{j}^{\geq J_\mathcal{G}}(k)) = J_\mathcal{G}$, and $d(X^{\geq J_\mathcal{G}}(k), \mathcal{G}^c) > \epsilon$. Now since $\mathbf{B}_k^\times$ did not occur, we must have $\sigma(\eta) > \widetilde{\tau}_k$ and $\widetilde{\tau}_k - T^{\geq J_\mathcal{G}}(k) \leq \rho(\epsilon)/\eta$, hence $\widetilde{\tau}_k - T_{k,1} \leq \frac{2l^*\hat{t}(\epsilon) + \rho(\epsilon)}{\eta}$. Therefore,

$$\big(\bigcap_{j=1}^{k-1}(\mathbf{A}_j^\times \cup \mathbf{B}_j^\times)^c \cap \mathbf{B}_j^\circ\big) \cap (\mathbf{A}_k^\times \cup \mathbf{B}_k^\times)^c \cap \{\mathbf{j}^{\geq J_\mathcal{G}}(k) < \infty, X^{\geq J_\mathcal{G}}(k) \in \mathcal{G}\} \subseteq \mathbf{B}_k^\circ.$$

With eq. (J.80) established, we can immediately get that

$$\big(\bigcap_{j=1}^{k-1}(\mathbf{A}_j^\times \cup \mathbf{B}_j^\times)^c \cap \mathbf{B}_j^\circ\big) \cap (\mathbf{A}_k^\times \cup \mathbf{B}_k^\times)^c = \big(\bigcap_{j=1}^{k-1}(\mathbf{A}_j^\times \cup \mathbf{B}_j^\times)^c \cap \mathbf{B}_j^\circ\big) \cap (\mathbf{A}_k^\times \cup \mathbf{B}_k^\times)^c \cap (\mathbf{A}_k^\circ \cup \mathbf{B}_k^\circ).$$

$$\text{(J.81)}$$

Next, recall the definitions of $\mathbf{A}_k^\circ$ in eq. (J.67) and $\mathbf{B}_k^\circ$ in eq. (J.68), and one can see that the two events $\mathbf{A}_k^\circ$ and $\mathbf{B}_k^\circ$ are mutually exclusive, since the former implies that the first exit occurs during the $k-$th cycle whereas the latter implies the first exit does not occur in the first $k$ cycles. This fact and eq. (J.81) allow us to conclude that

$$\bigcap_{i=1}^{k} \left(\mathbf{A}_i^\times \cup \mathbf{B}_i^\times \cup \mathbf{A}_i^\circ\right)^c = \bigcap_{i=1}^{k} \left(\mathbf{A}_i^\times \cup \mathbf{B}_i^\times\right)^c \cap \mathbf{B}_i^\circ = \left(\bigcap_{i=1}^{k-1} \left(\mathbf{A}_i^\times \cup \mathbf{B}_i^\times\right)^c \cap \mathbf{B}_i^\circ\right) \cap \left(\mathbf{A}_k^\times \cup \mathbf{B}_k^\times \cup \mathbf{A}_k^\circ\right)^c.$$
(J.82)

The last step is to use all the results so far to bound the probability of event

$$\mathbf{A}^\times(\epsilon, \delta, \eta) = \bigcup_{k \geq 1} \left(\bigcap_{i=1}^{k-1} \left(\mathbf{A}_i^\times \cup \mathbf{B}_i^\times \cup \mathbf{A}_i^\circ\right)^c\right) \cap \left(\mathbf{A}_k^\times \cup \mathbf{B}_k^\times\right).$$

Using eq. (J.82), we can see that (for any $x$ with $\|x\| \leq 2\epsilon$)

$$\mathbb{P}_x(\mathbf{A}^\times(\epsilon, \delta, \eta))$$

$$= \sum_{k \geq 1} \mathbb{P}_x\left(\left(\bigcap_{i=1}^{k-1} \left(\mathbf{A}_i^\times \cup \mathbf{B}_i^\times \cup \mathbf{A}_i^\circ\right)^c\right) \cap \left(\mathbf{A}_k^\times \cup \mathbf{B}_k^\times\right)\right)$$

$$= \sum_{k \geq 1} \mathbb{P}_x\left(\left(\bigcap_{i=1}^{k-1} \left(\mathbf{A}_i^\times \cup \mathbf{B}_i^\times\right)^c \cap \mathbf{B}_i^\circ\right) \cap \left(\mathbf{A}_k^\times \cup \mathbf{B}_k^\times\right)\right)$$

$$= \sum_{k \geq 1} \mathbb{P}_x\left(\mathbf{A}_k^\times \cup \mathbf{B}_k^\times \mid \bigcap_{i=1}^{k-1} \left(\mathbf{A}_i^\times \cup \mathbf{B}_i^\times\right)^c \cap \mathbf{B}_i^\circ\right) \cdot \prod_{j=1}^{k-1} \mathbb{P}_x\left(\bigcap_{i=1}^{j} \left(\mathbf{A}_i^\times \cup \mathbf{B}_i^\times\right)^c \cap \mathbf{B}_i^\circ \mid \bigcap_{i=1}^{j-1} \left(\mathbf{A}_i^\times \cup \mathbf{B}_i^\times\right)^c \cap \mathbf{B}_i^\circ\right)$$

$$= \sum_{k \geq 1} \mathbb{P}_x\left(\mathbf{A}_k^\times \cup \mathbf{B}_k^\times \mid \bigcap_{i=1}^{k-1} \left(\mathbf{A}_i^\times \cup \mathbf{B}_i^\times\right)^c \cap \mathbf{B}_i^\circ\right)$$
$$\cdot \prod_{j=1}^{k-1} \mathbb{P}_x\left(\left(\bigcap_{i=1}^{j-1} \left(\mathbf{A}_i^\times \cup \mathbf{B}_i^\times\right)^c \cap \mathbf{B}_i^\circ\right) \cap \left(\mathbf{A}_j^\times \cup \mathbf{B}_j^\times \cup \mathbf{A}_j^\circ\right)^c \mid \bigcap_{i=1}^{j-1} \left(\mathbf{A}_i^\times \cup \mathbf{B}_i^\times\right)^c \cap \mathbf{B}_i^\circ\right)$$

$$= \sum_{k \geq 1} \mathbb{P}_x\left(\mathbf{A}_k^\times \cup \mathbf{B}_k^\times \mid \bigcap_{i=1}^{k-1} \left(\mathbf{A}_i^\times \cup \mathbf{B}_i^\times\right)^c \cap \mathbf{B}_i^\circ\right) \cdot \prod_{j=1}^{k-1} \mathbb{P}_x\left(\left(\mathbf{A}_j^\times \cup \mathbf{B}_j^\times \cup \mathbf{A}_j^\circ\right)^c \mid \bigcap_{i=1}^{j-1} \left(\mathbf{A}_i^\times \cup \mathbf{B}_i^\times\right)^c \cap \mathbf{B}_i^\circ\right)$$

$$\leq \sum_{k \geq 1} \mathbb{P}_x\left(\mathbf{A}_k^\times \cup \mathbf{B}_k^\times \mid \bigcap_{i=1}^{k-1} \left(\mathbf{A}_i^\times \cup \mathbf{B}_i^\times\right)^c \cap \mathbf{B}_i^\circ\right) \cdot \prod_{j=1}^{k-1} \left(1 - \mathbb{P}_x\left(\mathbf{A}_j^\circ \mid \bigcap_{i=1}^{j-1} \left(\mathbf{A}_i^\times \cup \mathbf{B}_i^\times\right)^c \cap \mathbf{B}_i^\circ\right)\right).$$

This allows us to apply eq. (J.77)-eq. (J.79) and conclude that for all $\epsilon \in (0, \epsilon_7)$, all $\delta \in (0, \delta_7(\epsilon))$ and all $\eta > 0$

$$\sup_{|x| \leq 2\epsilon} \mathbb{P}_x(\mathbf{A}^\times(\epsilon, \delta, \eta)) \leq \sum_{k \geq 1} \left(\delta^{\alpha_1} \widetilde{\lambda}(\eta) \cdot \frac{c_* C}{2}\right) \cdot \left(1 - \delta^{\alpha_1} \widetilde{\lambda}(\eta) \cdot \frac{c_*}{2}\right)^{k-1}$$

$$= \frac{\delta^{\alpha_1} \widetilde{\lambda}(\eta) \cdot \frac{c_* C}{2}}{\delta^{\alpha_1} \widetilde{\lambda}(\eta) \cdot \frac{c_*}{2}} = C.$$

when $\eta$ is sufficiently small. □

Having established Lemma J.24, we are ready to provide a proof for Proposition J.23.

*Proof of Proposition J.23.* Define sets

$$E^+(z) \triangleq \{y \in \mathbb{R}^d : \, \boldsymbol{d}(y, \mathcal{G}) > (3 + 3\rho^*)z\}, \quad E^-(z) \triangleq \{y \in \mathbb{R}^d : \, \boldsymbol{d}(y, \mathcal{G}^c) < (3 + 3\rho^*)z\}$$

where $\rho^*$ is the fixed constant in Corollary J.7. From the continuity of measure, we have

$$\lim_{\epsilon \downarrow 0} \mu\big(h^{-1}(E^+(\epsilon))\big) = \mu\big(h^{-1}((\overline{\mathcal{G}})^c)\big), \quad \lim_{\epsilon \downarrow 0} \mu\big(h^{-1}(E^-(\epsilon))\big) = \mu\big(h^{-1}(\mathcal{G}^c)\big).$$

From Assumption I.7 we know $q \triangleq \mu\big(h^{-1}(\mathcal{G}^c)\big) > 0$. From Assumption I.9 we also have $\mu\big(h^{-1}(\partial\mathcal{G})\big) = 0$, hence $\mu\big(h^{-1}((\overline{\mathcal{G}})^c)\big) = q$ as well. Therefore, together with Lemma J.24, we know the existence of some $\epsilon_1 \in (0, \frac{\bar{\epsilon}}{3+3\rho^*})$ such that, for any $\epsilon \in (0, \epsilon_1)$, there is some $0 < \delta_1(\epsilon)$ with

$$\limsup_{\eta \downarrow 0} \sup_{\|x\| \leq 2\epsilon} \mathbb{P}_x(\mathbf{A}^\times(\epsilon, \delta, \eta)) < C.$$

and

$$(1 - C)q < \mu\big(h^{-1}(E^+(\epsilon))\big) \leq q \leq \mu\big(h^{-1}(E^-(\epsilon))\big) < (1 + C)q.$$

Henceforth, we fix some $\epsilon$ and $\delta$ in this range and prove the inequalities in the Proposition.

To proceed, let $J(\epsilon, \delta, \eta) \triangleq \sup\{k \geq 1 : \widetilde{\tau}_{k-1} < \sigma(\eta)\}$. We now show that on event $(\mathbf{A}^\times(\epsilon, \delta, \eta))^c$, we must have $J \leq J^\uparrow$ where (recall Definition J.1)

$$J^\uparrow(\epsilon, \delta, \eta) \triangleq \min\left\{k \geq 1 : Z_{T_{k,1}} \text{ is of type-}\big(E^+(\epsilon), \eta, \delta\big)\right\}$$

with $E^+(z) \triangleq \{y \in \mathbb{R}^d : \boldsymbol{d}(y, \mathcal{G}) > (3 + 3\rho^*)z\}$ where $\rho^*$ is the fixed constant in Corollary J.7. To see this via a proof of contradiction, let us assume that $J^\uparrow = j < J$ for some integer $j$. Now we make some observations.

- First of all, due to $J^\uparrow = j < J$ and eq. (J.17), on event $(\mathbf{A}^\times(\epsilon, \delta, \eta))^c \cap \{J^\uparrow = j < J\}$ we have

$$T_{j,i} - T_{j,i-1} \leq 2\bar{t}/\eta \; \forall i = 2, 3, \cdots, k^*;$$

- The definition of event $(\mathbf{A}^\times)^c$, especially the exclusion of event $\mathbf{A}^\times_{j,0}$ defined in eq. (J.58) now allows us to apply Lemma J.15 and show that

$$\sup_{s \in [0, T_{j,k^*} - T_{j,1}]} \left\| X^\eta_{\lfloor s \rfloor + T_{j,1}} - \boldsymbol{z}^\eta(s) \right\| < 2\epsilon.$$

Here

$$\boldsymbol{z}^\eta(s) = \widetilde{\boldsymbol{x}}^\eta\big(s, X^{(1)}; \boldsymbol{T}, \boldsymbol{W}\big) \; \forall s \geq 0$$

with $\boldsymbol{T} = \big(0, T_{j,2} - T_{j,1}, T_{j,3} - T_{j,1}, \cdots, T_{j,k^*} - T_{j,k^*-1}\big)$, $\boldsymbol{W} = \big(W_{j,1}, \cdots, W_{j,k^*}\big)$, and $X^{(1)} = X^\eta_{-1+T_{j,1}}$. Moreover, the previous bullet point allows us to apply Corollary J.7 and conclude that

$$\sup_{s \in [0, T_{j,k^*} - T_{j,1}]} \|\boldsymbol{z}^\eta(s) - \boldsymbol{z}^\eta_0(s)\| \leq 3\rho^*\epsilon$$

where $\boldsymbol{z}^\eta(s) = \widetilde{\boldsymbol{x}}^\eta\big(s, \boldsymbol{0}; \boldsymbol{T}, \boldsymbol{W}\big) \; \forall s \geq 0$ and $\rho^*$ is the fixed constant in Corollary J.7. As a result, $\left\| \boldsymbol{z}^\eta_0(T_{j,k^*} - T_{j,1}) - X^\eta_{T_{j,k^*}} \right\| < (2 + 3\rho^*)\epsilon$.

- However, due to $J^\uparrow = j$, we have $\boldsymbol{d}(\boldsymbol{z}^\eta_0(T_{j,k^*} - T_{j,1}), \mathcal{G}) > (3 + 3\rho^*)\epsilon$. This immediately implies that $X^\eta_{T_{j,k^*}} \notin \mathcal{G}$ and yields the contradiction $J \leq j$.

Similarly, one can show that on event $(\mathbf{A}^\times)^c$ we must have $J \geq J^\downarrow$ for

$$J^\downarrow(\epsilon, \delta, \eta) \triangleq \min\left\{k \geq 1 : Z_{T_{k,1}} \text{ is of type-}\big(E^-(\epsilon), \eta, \delta\big)\right\}$$

with $E^-(z) \triangleq \{y \in \mathbb{R}^d : \boldsymbol{d}(y, \mathcal{G}^c) < (3 + 3\rho^*)z\}$.

Besides, the following claim holds on event $(\mathbf{A}^\times)^c \cap \{J < \infty\}$.

- The definition of $(\mathbf{A}^\times)^c$ implies that during the $J-$th cycle the event $\mathbf{A}_J^\circ$ occurs whereas for any $j < J$ we have $\mathbf{B}_j^\circ$. Therefore, for any $k = 1, 2, \cdots, J$, we have

$$\widetilde{\tau}_k \wedge \sigma(\eta) - T_{k,1} \leq \frac{2l^* \hat{t}(\epsilon) + \rho(\epsilon)}{\eta}.$$

In particular, the coefficient $l^*$ on the R.H.S. can be justified as follows: by definition of $\mathbf{A}_k^\circ$ and $\mathbf{B}_k^\circ$, we can see that $\mathcal{J}_k^{\mathrm{cycle}}(\mathbf{j}_k) \leq J_\mathcal{G}$ for all $k \leq J$ (with $\mathbf{j}_J = k^*$ due to the occurrence of $\mathbf{A}_J^\circ$); the definition of $l^*$ in eq. (J.1) then implies that $l^* > \mathbf{j}_k$ for all $k \leq J$.

- Now we consider the following set

$$\mathbf{S}(\epsilon, \delta, \eta) \triangleq \cup_{k \geq 1} \left\{ \widetilde{\tau}_{k-1} + 1, \widetilde{\tau}_{k-1} + 2, \cdots, T_{k,1} - 1, T_{k,1} \right\}$$

the can be understood as the concatenation of all steps between any return time $\widetilde{\tau}_{k-1}$ and the first large jump time $T_{k,1}$ during the $k-$th cycle. Our discussion above implies that, on event $(\mathbf{A}^\times)^c$, we have then the discussion above have shown that, for

$$\min\{n \in \mathbf{S}(\epsilon, \delta, \eta) : \ Z_n \text{ is of type-}(E^+(\epsilon), \eta, \delta)\} \geq T_{J,1}.$$

It is worth noticing that the probability of $Z_1$ being of type-$(E^+(\epsilon), \eta, \delta)$ is (see the definition in eq. (J.47)) $H(\delta/\eta)p\big(E^+(\epsilon), \delta, \eta\big)$.

Therefore, for the following partition of the timeline

$$\mathbf{S}_{\mathrm{before}}(\epsilon, \delta, \eta) \triangleq \{n \in \mathbf{S}(\epsilon, \delta, \eta) : \ n \leq \sigma(\eta)\}, \ I_{\mathrm{before}}(\epsilon, \delta, \eta) \triangleq \#\mathbf{S}_{\mathrm{before}}(\epsilon, \delta, \eta),$$
$$\mathbf{S}_{\mathrm{after}}(\epsilon, \delta, \eta) \triangleq \{n \notin \mathbf{S}(\epsilon, \delta, \eta) : \ n \leq \sigma(\eta)\}, \ I_{\mathrm{after}}(\epsilon, \delta, \eta) \triangleq \#\mathbf{S}_{\mathrm{after}}(\epsilon, \delta, \eta),$$

we have $\sigma(\eta) = I_{\mathrm{before}} + I_{\mathrm{after}}$. Moreover, on event $(\mathbf{A}^\times)^c$, we must have

$$I_{\mathrm{after}} \leq J\big(2l^* \hat{t}(\epsilon) + \rho(\epsilon)\big)/\eta \leq J^\uparrow\big(2l^* \hat{t}(\epsilon) + \rho(\epsilon)\big)/\eta$$
$$I_{\mathrm{before}} \leq \min\{n \in \mathbf{S}(\epsilon, \delta, \eta) : \ Z_n \text{ is of type-}(E^+(\epsilon), \eta, \delta)\}\}.$$

Next, define geometric random variables with the following success rates

$$U_1(\epsilon, \delta, \eta) \sim \mathrm{Geom}\Big(H(\delta/\eta)p\big(E^+(\epsilon), \delta, \eta\big)\Big),$$
$$U_2(\epsilon, \delta, \eta) \sim \mathrm{Geom}\Big(p\big(E^+(\epsilon), \delta, \eta\big)\Big).$$

Given the results above for bounding $I_{\mathrm{before}}$ and $I_{\mathrm{after}}$ on event $(\mathbf{A}^\times)^c$, we have

$$\sup_{\|x\| \leq 2\epsilon} \mathbb{P}_x\Big(\lambda(\eta)\sigma(\eta) > u\Big)$$

$$\leq \sup_{\|x\| \leq 2\epsilon} \mathbb{P}_x\big((\mathbf{A}^\times)^c\big) + \sup_{\|x\| \leq 2\epsilon} \mathbb{P}_x\Big((\mathbf{A}^\times)^c \cap \big\{\lambda(\eta)(I_{\mathrm{before}} + I_{\mathrm{after}}) > u\big\}\Big)$$

$$\leq \sup_{\|x\| \leq 2\epsilon} \mathbb{P}_x\big((\mathbf{A}^\times)^c\big) + \sup_{\|x\| \leq 2\epsilon} \mathbb{P}_x\Big((\mathbf{A}^\times)^c \cap \big\{\lambda(\eta)I_{\mathrm{before}} > (1 - C)u\big\}\Big)$$

$$+ \sup_{\|x\| \leq 2\epsilon} \mathbb{P}_x\Big((\mathbf{A}^\times)^c \cap \big\{\lambda(\eta)I_{\mathrm{after}} > Cu\big\}\Big)$$

$$\leq \sup_{\|x\| \leq 2\epsilon} \mathbb{P}_x\big((\mathbf{A}^\times)^c\big) + \underbrace{\mathbb{P}\big(\lambda(\eta) \cdot U_1 > (1 - C)u\big)}_{\triangleq (\mathrm{I})} + \underbrace{\mathbb{P}\Big(\lambda(\eta) \cdot U_2 \cdot \frac{2l^* \hat{t}(\epsilon) + \rho(\epsilon)}{\eta} > Cu\Big)}_{\triangleq (\mathrm{II})}.$$

For term (I), recall that $\lambda(\eta) = H(1/\eta)\widetilde{\lambda}(\eta)$. Corollary J.14 and the regularly varying nature of $H(\cdot)$ then imply that

$$\lim_{\eta \downarrow 0} \frac{H(\delta/\eta)p\big(E^+(\epsilon), \delta, \eta\big)}{\lambda(\eta)} = \frac{H(\delta/\eta) \cdot p\big(E^+(\epsilon), \delta, \eta\big)}{\frac{H(1/\eta)}{\delta^{\alpha_1}} \cdot \delta^{\alpha_1}\widetilde{\lambda}(\eta)} = \mu\big(h^{-1}(E^+(\epsilon))\big) > (1 - C)q.$$

From Lemma G.3, we then have $\limsup_{\eta\downarrow 0} (\mathrm{I}) \leq \exp(-(1-C)^2 u)$. For term (II), let $C(\epsilon) \triangleq 2l^*\hat{t}(\epsilon) + \rho(\epsilon)$ and note that

$$(\mathrm{II}) = \mathbb{P}\Big(U_2 > \frac{C}{C(\epsilon)} \cdot \frac{\widetilde{\lambda}(\eta)}{\lambda(\eta)/\eta} \cdot \frac{1}{\widetilde{\lambda}(\eta)}\Big).$$

Moreover, since

$$\frac{\widetilde{\lambda}(\eta)}{\lambda(\eta)/\eta} \in RV_{-\alpha_1+1}(\eta),$$

we know that for any $M > 0$, we have $\frac{\widetilde{\lambda}(\eta)}{\lambda(\eta)/\eta} > M$ eventually as $\eta \downarrow 0$. Therefore, given any $M > 0$, we have $(\mathrm{II}) \leq \mathbb{P}\Big(U_2 > M/\widetilde{\lambda}(\eta)\Big)$ for any $\eta$ sufficiently small. Now from Lemma G.3, $\limsup_{\eta\downarrow 0} (\mathrm{II}) \leq \exp(-M)$ for any $M > 0$, and here we fix some $M > 0$ such that $\exp(-M) < C$. In summary, we have shown

$$\limsup_{\eta\downarrow 0} \sup_{\|x\|\leq 2\epsilon} \mathbb{P}_x\Big(\lambda(\eta)\sigma(\eta) > u\Big) \leq 2C + \exp(-(1-C)^2 qu)$$

and established the upper bound. The lower bound can be shown by an almost identical approach. In particular, since

$$\inf_{\|x\|\leq 2\epsilon} \mathbb{P}_x\Big(\lambda(\eta)\sigma(\eta) > u\Big) \geq \inf_{\|x\|\leq 2\epsilon} \mathbb{P}_x\Big((\mathbf{A}^\times)^c \cap \big\{\lambda(\eta)(I_{\text{before}} + I_{\text{after}}) > u\big\}\Big)$$

$$\geq \inf_{\|x\|\leq 2\epsilon} \mathbb{P}_x\Big((\mathbf{A}^\times)^c \cap \big\{\lambda(\eta)I_{\text{before}} > u\big\}\Big)$$

$$\geq \mathbb{P}\big(\lambda(\eta)\cdot U_1' > u\big) - \sup_{\|x\|\leq 2\epsilon} \mathbb{P}_x\Big((\mathbf{A}^\times)^c\Big)$$

where $U_1'(\epsilon, \delta, \eta) \sim \mathrm{Geom}\Big(H(\delta/\eta)p\big(E^-(\epsilon), \delta, \eta\big)\Big)$. The same calculation above for term (I) can be used here to provide a lower bound for $\liminf_{\eta\downarrow 0} \mathbb{P}\big(\lambda(\eta)\cdot U_1' > u\big)$ and conclude the proof. $\square$

Now we are ready to prove Theorem I.1, the main theorem of this section.

*Proof of Theorem I.1.* Recall that our choice of scaling function $\lambda(\cdot)$ in eq. (J.56) is regularly varying (w.r.t. $\eta$) with index $1 + J_{\mathcal{G}}$. Fix some $x \in \mathcal{G}$, $t > 0$ and $C \in (0, 1)$. It suffices to show that

$$\limsup_{\eta\downarrow 0} \mathbb{P}_x(\sigma(\eta)\lambda(\eta) > t) \leq \exp(-q(1-C)^3 t) + 2C,$$

$$\liminf_{\eta\downarrow 0} \mathbb{P}_x(\sigma(\eta)\lambda(\eta) > t) \geq \exp(-q(1+C)t) - C.$$

First, we are able to pick some $\epsilon \in (0, 1)$ sufficiently small such that $\boldsymbol{d}(x, \mathcal{G}^c) > \epsilon$ and Proposition J.23 is applicable. Due to Lemma J.11, for event

$$A \triangleq \{T_{\text{return}}(\eta, \epsilon) \leq \rho(\epsilon)/\eta,\ X_n^\eta \in \mathcal{G}\ \forall n \leq T_{\text{return}}(\eta, \epsilon)\},$$

we have $\lim_{\eta\downarrow 0} \mathbb{P}_x(A) = 1$. Next, on event $A$, we have $\sigma(\eta) - T_{\text{return}}(\eta, \epsilon) > 0$ and $\Big\|X_{T_{\text{return}}(\eta,\epsilon)}^\eta\Big\| \leq 2\epsilon$. Moreover, by combining Proposition J.23 with the strong Markov property at stopping time $T_{\text{return}}(\eta, \epsilon)$, we have

$$\limsup_{\eta\downarrow 0} \mathbb{P}_x\Big(\big(\sigma(\eta) - T_{\text{return}}(\eta, \epsilon)\big)\lambda(\eta) > (1-C)t \mid A\Big) \leq 2C + \exp(-(1-C)^3 qt),$$

$$\liminf_{\eta\downarrow 0} \mathbb{P}_x\Big(\big(\sigma(\eta) - T_{\text{return}}(\eta, \epsilon)\big)\lambda(\eta) > t \mid A\Big) \geq -C + \exp(-(1+C)qt).$$

Observe that

$$\mathbb{P}_x(\sigma(\eta)\lambda(\eta) > t) \leq \mathbb{P}_x(\{\sigma(\eta)\lambda(\eta) > t\} \cap A) + \mathbb{P}_x(A^c)$$

$$\leq \mathbb{P}_x\Big(\big(\sigma(\eta) - T_{\text{return}}(\eta, \epsilon)\big)\lambda(\eta) > (1-C)t \mid A\Big)\mathbb{P}_x(A)$$

$$+ \mathbb{P}_x\Big(\{T_{\text{return}}(\eta, \epsilon)\lambda(\eta) > Ct\} \cap A\Big) + \mathbb{P}_x(A^c).$$

Besides, on event $A$ we have $T_{\text{return}}(\eta, \epsilon) \le O(1/\eta)$ as $\eta \downarrow 0$. Given that $\lambda(\eta) \in RV_{(1+J_{\mathcal{G}})}(\eta)$, we have $T_{\text{return}}(\eta, \epsilon)\lambda(\eta) \le Ct$ on event $A$ for all $\eta$ sufficiently small. Therefore, by applying the bounds above, we establish that $\limsup_{\eta \downarrow 0} \mathbb{P}_x(\sigma(\eta)\lambda(\eta) > t) \le 2C + \exp(-(1-C)^3 qt)$. Similarly, in order to show the lower bound, observe that

$$\mathbb{P}_x(\sigma(\eta)\lambda(\eta) > t) \ge \mathbb{P}_x(\{\sigma(\eta)\lambda(\eta) > t\} \cap A)$$
$$\ge \mathbb{P}_x\Big(\big(\sigma(\eta) - T_{\text{return}}(\eta, \epsilon)\big)\lambda(\eta) > t \mid A\Big)\mathbb{P}_x(A).$$

Taking $\liminf$ on both sides yields $\liminf_\eta \mathbb{P}_x(\sigma(\eta)\lambda(\eta) > t) \ge -C + \exp(-(1+C)qt)$ and concludes the proof. $\square$

## K  PROOF OF TECHNICAL LEMMAS

*Proof of Lemma G.3.*  For any $\epsilon > 0$,

$$\mathbb{P}\Big(U(\epsilon) > \frac{1}{b(\epsilon)}\Big) = \Big(1 - a(\epsilon)\Big)^{\lfloor 1/b(\epsilon) \rfloor}.$$

By taking logarithm on both sides, we have

$$\ln \mathbb{P}\Big(U(\epsilon) > \frac{1}{b(\epsilon)}\Big) = \lfloor 1/b(\epsilon) \rfloor \ln\Big(1 - a(\epsilon)\Big)$$

$$= \frac{\lfloor 1/b(\epsilon) \rfloor}{1/b(\epsilon)} \frac{\ln\Big(1 - a(\epsilon)\Big)}{-a(\epsilon)} \frac{-a(\epsilon)}{b(\epsilon)}.$$

Since $\lim_{x \to 0} \frac{\ln(1+x)}{x} = 1$, we know that for $\epsilon$ sufficiently small, we will have

$$-c\frac{a(\epsilon)}{b(\epsilon)} \le \ln \mathbb{P}\Big(U(\epsilon) > \frac{1}{b(\epsilon)}\Big) \le -\frac{a(\epsilon)}{c \cdot b(\epsilon)}. \tag{K.1}$$

By taking exponential on both sides, we conclude the proof. $\square$

*Proof of Lemma G.4.*  To begin with, for any $\epsilon > 0$ we have

$$\mathbb{P}\Big(U(\epsilon) \le \frac{1}{b(\epsilon)}\Big) = 1 - \mathbb{P}\Big(U(\epsilon) > \frac{1}{b(\epsilon)}\Big).$$

Using bound eq. (K.1), we know that for $\epsilon$ sufficiently small, $\mathbb{P}(U(\epsilon) > 1/b(\epsilon)) \ge \exp(-c \cdot a(\epsilon)/b(\epsilon))$. The upper bound follows from the generic bound $1 - \exp(-x) \le x$, $\forall x \in \mathbb{R}$ with $x = c \cdot a(\epsilon)/b(\epsilon)$.

Now we move onto the lower bound. Again, from bound eq. (K.1), we know that for sufficiently small $\epsilon$, we will have

$$\mathbb{P}\Big(U(\epsilon) \le \frac{1}{b(\epsilon)}\Big) \ge 1 - \exp(-\frac{1}{\sqrt{c}} \cdot \frac{a(\epsilon)}{b(\epsilon)}).$$

Due to the assumption that $\lim_{\epsilon \downarrow 0} a(\epsilon)/b(\epsilon) = 0$ and the fact that $1 - \exp(-x) \ge \frac{x}{\sqrt{c}}$ for $x > 0$ sufficiently close to $0$, we will have (for $\epsilon$ small enough) $\mathbb{P}\Big(U(\epsilon) \le \frac{1}{b(\epsilon)}\Big) \ge \frac{1}{c} \cdot \frac{a(\epsilon)}{b(\epsilon)}$. $\square$

*Proof of Lemma G.5.*  Let $a_k \triangleq x_k - \widetilde{x}_k$. Using intermediate value theorem, one can easily see that

$$a_k = \eta \sum_{j=1}^k \Big(\nabla g(\widetilde{x}_{j-1}) - \nabla g(x_{j-1})\Big) + \eta(z_1 + \cdots + z_k) + x - \widetilde{x};$$

$$\Rightarrow \|a_k\| \le \eta C(\|a_0\| + \cdots \|a_{k-1}\|) + \widetilde{c}.$$

The desired bound then follows immediately from Gronwall's inequality. $\square$

*Proof of Lemma J.1.* Due to the finiteness of all types $\boldsymbol{j}'$ with $\mathcal{J}_{\text{type}}(\boldsymbol{j}') < J_{\mathcal{G}}$, it suffices to fix one of such $\boldsymbol{j}'$ and prove the existence of constants $\bar{t}, \bar{\delta}, \epsilon_0$.

Let $\bar{\epsilon}$ be the constants in eq. (J.3). For any $r \in (0, \bar{\epsilon})$, define the following stopping time

$$\tau_r(x) \triangleq \min\{t \geq 0 : \boldsymbol{x}_t(x) \in \overline{B(\boldsymbol{0}, r)}\}.$$

Due to Assumptions I.4 and I.5 as well as continuity in Poincaré Map (see Theorem 12 in Immler & Traut (2019)), for any fixed $r > 0$ we know that $\tau_r(\cdot)$ is a continuous function on $\overline{\mathcal{G}}$. Note that, by definition, we have $\tau_r(x) = 0$ for any $x \in \overline{B(\boldsymbol{0}, r)}$. Due to $\overline{\mathcal{G}}$ being compact and eq. (J.3), we know the existence of some constant $T_r \in (0, \infty)$ such that $\sup_{x \in \bar{\mathcal{G}}} \tau_r(x) \leq T_r$ and $\boldsymbol{x}_t(x) \in \overline{B(\boldsymbol{0}, r)}$ for any $x \in \overline{\mathcal{G}}, t \geq T_r$.

Due to Assumption I.8 and the finiteness of all types $\boldsymbol{j}'$ with $\mathcal{J}_{\text{type}}(\boldsymbol{j}') < J_{\mathcal{G}}$, there exists a $\epsilon_0 \in (0, \bar{\epsilon}/2)$ such that

$$\sup_{s \geq 0} \boldsymbol{d}\big(\widetilde{\boldsymbol{x}}(s, \boldsymbol{0}; \boldsymbol{t}, \boldsymbol{w}), \mathcal{G}^c\big) > 2\epsilon_0 \tag{K.2}$$

for all $\boldsymbol{j}'$ with $\mathcal{J}_{\text{type}}(\boldsymbol{j}') < J_{\mathcal{G}}, \boldsymbol{t} \in \{0\} \times \mathbb{R}_+^{|\boldsymbol{j}'|-1}, \boldsymbol{w} \in \mathcal{A}_{\boldsymbol{j}'}^{\text{type}}$. We fix such $\epsilon_0 > 0$. Here is one implication that is worth mentioning. Recall that $\boldsymbol{j}$ is fixed in the description of the Lemma and $k' = |\boldsymbol{j}|$. If

$$\inf_{s \geq 0} \boldsymbol{d}\big(\widetilde{\boldsymbol{x}}(s, \boldsymbol{0}; \boldsymbol{t}^{(k')}, \boldsymbol{w}^{(k')}), \mathcal{G}^c\big) \leq \epsilon_0$$

for some $\boldsymbol{t}^{(k')} \in \{0\} \times \mathbb{R}_+^{k'-1}, \boldsymbol{w}^{(t')} \in \mathcal{A}_{\boldsymbol{j}}^{\text{type}}$, then we must have

$$\boldsymbol{d}\big(\widetilde{\boldsymbol{x}}(\sum_{i=1}^{k'} t_i, \boldsymbol{0}; \boldsymbol{t}^{(k')}, \boldsymbol{w}^{(k')}), \mathcal{G}^c\big) \leq \epsilon_0.$$

To show the existence of some $\bar{\delta} > 0$ we appeal to a proof by contradiction. (For a clean presentation, in $\boldsymbol{t}$ and $\boldsymbol{w}$ we omit the $(k')$ term in the superscript since the cardinality is fixed.) Assume that we can find a sequence $\big(\boldsymbol{t}^n, \boldsymbol{w}^n\big)_{n \geq 1}$ such that $\min\big\{\|w_1^n\|, \cdots, \|w_{k'}^n\|\big\} \leq 1/n$ and $\boldsymbol{d}\big(\widetilde{\boldsymbol{x}}\big(\sum_{i=1}^{k'} t_i^n, \boldsymbol{0}; \boldsymbol{t}^n, \boldsymbol{w}^n\big), \mathcal{G}^c\big) \leq \epsilon_0$ for any $n$. Due to the truncation operator $\varphi_b(\cdot)$ in the definition of $\widetilde{\boldsymbol{x}}_t$, without loss of generality we can replace all jumps $w_j^n$ by $\varphi_b\big(w_j^n\big)$ to ensure that $w_j^n$ is always in a compact set. Define

$$\boldsymbol{y}_j^n = \widetilde{\boldsymbol{x}}\big(t_j^n, \boldsymbol{0}; \boldsymbol{t}^n, \boldsymbol{w}^n\big) \ \forall j = 1, 2, \cdots, k'.$$

By picking a subsequence when necessary, we can further assume that

- For any $i \geq 1$, $w_j^{(n)}$ converges to some $w_i^*$ and $\boldsymbol{y}_i^n$ converges to some $\boldsymbol{y}_i^*$. In particular, there exists some $I \in [k']$ such that $w_I^* = \boldsymbol{0}$.

- Also, since $\mathcal{G}^c$ is a closed set, we have $\boldsymbol{d}(\boldsymbol{y}_{k'}^*, \mathcal{G}^c) \leq \epsilon_0$.

- For any $i = 2, 3, \cdots, k'$, $t_i^n$ either converges to some finite $t_i^*$, or $\lim_n t_i^n = \infty$.

- Note that

$$\lim_n t_i^n = \infty \Rightarrow \boldsymbol{y}_i^* = \boldsymbol{0} \tag{K.3}$$

  for the following reason: for any $r \in (0, \bar{\epsilon})$, our discussion about $\tau_r(x)$ at the beginning of the proof implies that $\limsup_n \|\boldsymbol{y}_i^n\| \leq r$.

Let $\widetilde{I}$ be the largest index $i \in [k']$ such that $\boldsymbol{y}_i^* = \boldsymbol{0}$. In case that we cannot find such index, let $\widetilde{I} = 0$. We first consider the case where $\widetilde{I} = 0$. The bullet points above then imply that, for any $i \geq 1$, we must have $\lim_n t_i^n = t_i^* < \infty$. Now due to boundedness of $t_i^*$ and the continuity of ODE flow, for

$$\boldsymbol{t}^* \triangleq (0, t_2^*, t_2^*, \cdots, t_{k'}^*),$$
$$\boldsymbol{w}^* \triangleq (w_1^*, w_2^*, w_3^*, \cdots, w_{k'}^*),$$

we have

$$\widetilde{\boldsymbol{x}}(\sum_{i=1}^{k'} t_i^*, \boldsymbol{0}; \boldsymbol{t}^*, \boldsymbol{w}^*) = \boldsymbol{y}_{k'}^*.$$

Now recall that $w_I^* = 0$ for some $I \in [k']$. By removing the vacuous (size-zero) jumps in $(\boldsymbol{t}^*, \boldsymbol{w}^*)$, we now know the existence of some $\widetilde{k}' \in \mathbb{N}$, $\widetilde{\boldsymbol{w}}^* \in (\mathbb{R}^d \backslash \mathbb{0})^{\widetilde{k}'}$, $\widetilde{\boldsymbol{t}}^* \in \{0\} \times \mathbb{R}_+^{\widetilde{k}'}$ such that $\boldsymbol{d}\big(\widetilde{\boldsymbol{x}}(\sum_{i=1}^{\widetilde{k}'} \widetilde{t}_i^*, \boldsymbol{0}; \widetilde{\boldsymbol{t}}^*, \widetilde{\boldsymbol{w}}^*), \mathcal{G}^c\big) \leq \epsilon_0$. Meanwhile, the condition $\mathcal{J}_{\text{type}}^\downarrow(\boldsymbol{j}) < J_{\mathcal{G}}$ implies that, for the total cost of jumps in $\widetilde{\boldsymbol{w}}^*$ after removal of size-zero jumps, $\sum_{i=1}^{\widetilde{k}'} J(\widetilde{\boldsymbol{w}}_j^*) < \mathcal{J}_G$. However, this contradicts eq. (K.2).

Similarly, in the case with $\widetilde{I} \geq 1$, we have that for any $i > \widetilde{I}$, we must have $\lim_n t_i^n = t_i^* < \infty$. From the boundedness of $t_i^*$ and the continuity of ODE flow, for

$$\boldsymbol{t}^* \triangleq (0, t_{\widetilde{I}+1}^*, t_{\widetilde{I}+2}^*, \cdots, t_{k'}^*), \tag{K.4}$$

$$\boldsymbol{w}^* \triangleq (w_{\widetilde{I}}^*, w_{\widetilde{I}+1}^*, w_{\widetilde{I}+2}^*, \cdots, w_{k'}^*), \tag{K.5}$$

we have

$$\widetilde{\boldsymbol{x}}(\sum_{i=\widetilde{I}+1}^{k'} t_i^*, \boldsymbol{0}; \boldsymbol{t}^*, \boldsymbol{w}^*) = \boldsymbol{y}_{k'}^*. \tag{K.6}$$

In particular, if $\widetilde{I} = 1$, then for the index $I$ with $w_I^* = \boldsymbol{0}$, we must have that $I \geq \widetilde{I}$, meaning that there is at least one vacuous jump in $\boldsymbol{w}^*$. The same argument for the case $\widetilde{I} = 0$ above can lead to the same contradiction with eq. (K.2). Otherwise, with $\widetilde{I} \geq 2$, we already know that the accumulated cost of all jumps in $w^*$ is strictly less than $J_{\mathcal{G}}$, yet we still have $\boldsymbol{d}\big(\widetilde{\boldsymbol{x}}(\sum_{i=\widetilde{I}+1}^{k'} t_i^*, \boldsymbol{0}; \boldsymbol{t}^*, \boldsymbol{w}^*), \mathcal{G}^c\big) \leq \epsilon_0$. This contradicts eq. (K.2) again.

In summary, we have established the existence of the lower bound $\bar{\delta} > 0$ on jump sizes. We fix such $\bar{\delta} > 0$. The existence of $\bar{t} < \infty$ can be shown by an almost identical argument. In particular, if such $\bar{t} < \infty$ does not exist, then by picking a subsequence if needed we are able to find a converging sequence $(\boldsymbol{t}^n, \boldsymbol{w}^n)_{n \geq 1}$ such that eq. (K.6) holds and $\widetilde{I} \geq 1$ due to inter-arrival time blowing up to infinity. In particular, since $\boldsymbol{y}_1^* = w_1^*$ and $\|w_1^*\| \geq \bar{\delta} > 0$, we must have $\widetilde{I} \geq 2$. By considering the same $\boldsymbol{t}^*, \boldsymbol{w}^*$ pair in equation K.4 and equation K.5, one can see that the accumulated cost of jumps in $\boldsymbol{w}$ is strictly less than $J_{\mathcal{G}}$ and yield a contradiction with eq. (K.2). $\qquad \square$

*Proof of Lemma J.2.* Due to the finiteness of all types $\boldsymbol{j}'$ with $\mathcal{J}_{\text{type}}(\boldsymbol{j}') < J_{\mathcal{G}}$, it suffices to fix one of such $\boldsymbol{j}'$ and prove the existence of the positive constant $\epsilon_1$.

To proceed with a proof by contradiction, the assume that such $\epsilon_1 > 0$ does not exists. (For a clean presentation, in $\boldsymbol{t}$ and $\boldsymbol{w}$ we omit the $(k')$ term in the superscript since the cardinality is fixed.) As a result, we are able to pick a sequence $(x^n, \boldsymbol{t}^n, \boldsymbol{w}^n)_{n \geq 1}$ such that one of the following two cases must occur:

- $\lim_n \|x^n\| = 0$; For any $n \geq 1$, $\inf_{s \geq 0} \boldsymbol{d}\big(\widetilde{\boldsymbol{x}}(s, x^n; \boldsymbol{t}^n, \boldsymbol{w}^n), \mathcal{G}^c\big) \leq \epsilon_0/2$ and there is some $j_n \in [k']$ such that $\boldsymbol{t}_{j_n}^n \geq 2\bar{t}$;

- $\lim_n \|x^n\| = 0$; For any $n \geq 1$, $\inf_{s \geq 0} \boldsymbol{d}\big(\widetilde{\boldsymbol{x}}(s, x^n; \boldsymbol{t}^n, \boldsymbol{w}^n), \mathcal{G}^c\big) \leq \epsilon_0/2$ and there is some $j_n \in [k']$ such that $\|\boldsymbol{w}_{j_n}^n\| \leq \bar{\delta}/2$.

We detail the analysis for the first case, as the second case can be addressed by an almost identical argument. First of all, due to the truncation operator $\varphi_b(\cdot)$ in the definition of $\widetilde{\boldsymbol{x}}_t$, without loss of generality we can replace all jumps $w_j^n$ by $\varphi_b(w_j^n)$ to ensure that $w_j^n$ is always in a compact set. Define

$$\boldsymbol{y}_j^n = \widetilde{\boldsymbol{x}}\big(t_j^n, x^n; \boldsymbol{t}^n, \boldsymbol{w}^n\big) \ \forall j = 1, 2, \cdots, k'.$$

By picking a subsequence if necessary, we can further assume that

- For any $i \geq 1$, $w_j^{(n)}$ converges to some $w_i^*$ and $\boldsymbol{y}_i^n$ converges to some $\boldsymbol{y}_i^*$.

- Also, since $\mathcal{G}^c$ is a closed set, we have $\boldsymbol{d}(\boldsymbol{y}_{k'}^*, \mathcal{G}^c) \leq \epsilon_0/2$.

- For any $i = 2, 3, \cdots, k'$, $t_i^n$ either converges to some finite $t_i^*$, or $\lim_n t_i^n = \infty$. In particular, there is some $I \in [k']$ such that $\lim_n t_I^n \geq 2\bar{t}$.

- Due to eq. (K.3), $\lim_n t_i^n = \infty$ would imply $\boldsymbol{y}_i^* = \boldsymbol{0}$.

Let $I_1 = \max\{i \in [k'] : \lim_n t_i^n \in [2\bar{t}, \infty)\}$ and $I_2 = \max\{i \in [k'] : \lim_n t_i^n = \infty\}$. If either of the two sets above is empty, let the corresponding $I_1$ or $I_2$ be 0. The discussion above implies that at least one of them must be non-zero, so there are only two possibilities: (i) $0 \leq I_1 < I_2$; (ii) $0 \leq I_2 < I_1$. We consider each scenario respectively.

First, if $0 \leq I_1 < I_2$, then $\lim_n y_{I_2}^n = y_{I_2}^* = \boldsymbol{0}$, implying that $I_2 < k'$. Moreover, for any $i = I_2 + 1, I_2 + 2, \cdots, k'$, $\lim_n t_i^n = t_i^* < \infty$. Using the boundedness of $t_i^*$ and the continuity of ODE flow, for

$$\boldsymbol{t}^* \triangleq (0, t_{1+I_2}^*, t_{2+I_2}^*, \cdots, t_{k'}^*), \quad \boldsymbol{w}^* \triangleq (w_{I_2}^*, w_{1+I_2}^*, w_{2+I_2}^*, \cdots, w_{k'}^*),$$

we have $\widetilde{\boldsymbol{x}}(\sum_{i=1+I_2}^{k'} t_i^*, \boldsymbol{0}; \boldsymbol{t}^*, \boldsymbol{w}^*) = \boldsymbol{y}_{k'}^*$ with $\boldsymbol{d}(\boldsymbol{y}_{k'}^*, \mathcal{G}^c) \leq \epsilon_0/2$. However, this contradicts equation J.2.

Next, in scenario (ii) with $0 \leq I_2 < I_1$, we have $\lim_n t_{I_1}^n = t_{I_1}^* \geq 2\bar{t}$. Moreover, for any $i = I_2 + 1, I_2 + 2, \cdots, k'$, $\lim_n t_i^n = t_i^* < \infty$. Using the boundedness of $t_i^*$ and the continuity of ODE flow, for

$$\boldsymbol{t}^* \triangleq (0, t_{1+I_2}^*, t_{2+I_2}^*, \cdots, t_{k'}^*), \quad \boldsymbol{w}^* \triangleq (w_{I_2}^*, w_{1+I_2}^*, w_{2+I_2}^*, \cdots, w_{k'}^*),$$

we have $\widetilde{\boldsymbol{x}}(\sum_{i=1+I_2}^{k'} t_i^*, \boldsymbol{0}; \boldsymbol{t}^*, \boldsymbol{w}^*) = \boldsymbol{y}_{k'}^*$ with $\boldsymbol{d}(\boldsymbol{y}_{k'}^*, \mathcal{G}^c) \leq \epsilon_0/2$. However, $bmw^*$ is still of type $\boldsymbol{j}$ with $\mathcal{J}_{\text{type}}^{\downarrow}(\boldsymbol{j}) < J_{\mathcal{G}}$, and there is some $i$ such that $\boldsymbol{t}_i^* \geq 2\bar{t}$. This would contradict Lemma J.1.

In summary, we have established the existence of $\epsilon_1 > 0$ such that $t_j < 2\bar{t}$ is a necessary condition for any $x \in B(\boldsymbol{0}, \epsilon_1)$ and any $\boldsymbol{t} = (t_1, \cdots, t_{k'}) \in \{0\} \times \mathbb{R}_+^{k'-1}$, $\boldsymbol{w} = (w_1, \cdots, w_{k'}) \in \mathcal{A}_{\boldsymbol{j}}^{\text{type}}$ with $\inf_{s \geq 0} \boldsymbol{d}(\widetilde{\boldsymbol{x}}(s, x; \boldsymbol{t}^{(k)}, \boldsymbol{w}^{(k)}), \mathcal{G}^c) \leq \epsilon_0/2$. As mentioned above, the necessity of $\|w_j\| > \bar{\delta}/2$ can be shown in an almost identical way. We omit the details here and conclude the proof. $\square$

*Proof of Lemma J.3.* Since there are only finitely many $(i_1, \cdots, i_m)$ with $\mathcal{J}(i_1, \cdots, i_m) < J_{\mathcal{G}}$, it suffices to fix one of such $(i_1, \cdots, i_m)$ and establish the existence of the required $\epsilon_0, \delta_0$. (Henceforth, let $k = \sum_{j=1} i_m$.)

Assumption I.8, together with the bound in eq. (J.6), implies the existence of some $\epsilon_1 \in (0, \bar{\epsilon})$ such that

$$\sup_{t \geq 0} \boldsymbol{d}(\widetilde{\boldsymbol{x}}(s, \boldsymbol{0}; \boldsymbol{t}^{(k)}, \boldsymbol{w}^{(k)}) > \epsilon_1 \tag{K.7}$$

for any $\boldsymbol{w}^{(k)} \in \mathcal{A}(i_1, \cdots, i_m)$, $\boldsymbol{t}^{(k)} \in \{0\} \times \mathbb{R}_+^{k-1}$. For $\epsilon_0 = \epsilon_1/2$, we establish the existence of the prescribed $\delta_0$ via a proof by contradiction. (Henceforth we drop the notational dependence on $(k)$ when referencing sequences $\boldsymbol{t}^{(k)}, \boldsymbol{w}^{(k)}$ since the cardinality $k$ is fixed.)

Assume the existence of a sequence $(x_n)_{n \geq 1}$ in $\mathbb{R}^d$, a sequence of real positives $(s_n)_{n \geq 1}$, a sequence $(\boldsymbol{t}_n)_{n \geq 1} = (t_{1,n}, \cdots, t_{k,n})_{n \geq 1}$ in $\{0\} \times \mathbb{R}_+^{k-1}$, and a sequence $(\boldsymbol{w}_n)_{n \geq 1} = (w_{1,n}, \cdots, w_{k,n})_{n \geq 1}$ in $\mathcal{A}(i_1, \cdots, i_m)$ such that $\lim_n \|x_n\| = 0$ and

$$\boldsymbol{d}(\widetilde{\boldsymbol{x}}(s_n, x_n; \boldsymbol{t}_n, \boldsymbol{w}_n), \mathcal{G}^c) \leq \epsilon_0.$$

Due to existence of the clipping operator, all jumps $w_{j,n}$ can be replaced by $\varphi_b(w_{j,n})$ without loss of generality to ensure that all $w_{j,n}$ are in a compact set. Also, without loss of generality, all $s_n$ can be chosen as

$$s_n = \inf\{s \geq 0 : \boldsymbol{d}(\widetilde{\boldsymbol{x}}(s, x_n; \boldsymbol{t}_n, \boldsymbol{w}_n), \mathcal{G}^c) \leq \epsilon_0\}.$$

From Assumption eq. (J.6), one can see that $s_n$ must be equal to $\sum_{i=1}^{j_n} t_{i,n}$ for some $j_n \in [k]$, i.e., it must be the arrival time of some jump. Moreover, one can easily see that for
$$y_n = \widetilde{\boldsymbol{x}}(s_n, x_n; \, \boldsymbol{t}_n, \boldsymbol{w}_n),$$
we always have $\boldsymbol{d}(y_n, \mathcal{G}) \le b$ so all $y_n$ are in a compact set as well due to the boundedness of $\mathcal{G}$. Therefore, by picking a subsequence when necessary, we can further assume that

- There exists some $j^* \in [k]$ such that $s_n = \sum_{i=1}^{j^*} t_{i,n}$ for all $n \ge 1$, i.e. $s_n$ is always the arrival time of the $j^*$−th jump;

- For any $j \in [j^*]$, there exists some $y_j^*$ such that for $y_{n,j} = \widetilde{\boldsymbol{x}}(\sum_{i=1}^{j} t_{i,n}, x_n; \, \boldsymbol{t}_n, \boldsymbol{w}_n)$ we have $\lim_n y_{n,j} = y_j^*$; In particular, for any $j < j^*$ we have $\boldsymbol{d}(y_j^*, \mathcal{G}^c) > \epsilon_1$ and $\boldsymbol{d}(y_{j^*}^*, \mathcal{G}^c) \le \epsilon_0$;

- For any $j \in [k]$, there exists some $w_j^*$ such that $\lim_n w_{j,n} = w_j^*$; Moreover, note that $y_1^* = w_1^*$.

- In particular, $(w_1^*, \cdots, w_k^*) \in \mathcal{A}(i_1, \cdots, i_m)$;

- For any $j \in [k]$, either there exists some $t_j^* < \infty$ such that $\lim_n t_{j,n} = t_j^*$, or $\lim_n t_{j,n} = \infty$ (in this case we let $t_j^* = \infty$); in the latter case, due to the same argument in eq. (K.3), we must have $\lim_n \widetilde{\boldsymbol{x}}(\sum_{i=1}^{j} t_{i,n}, x_i; \, \boldsymbol{t}_n, \boldsymbol{w}_n) = \boldsymbol{0}$;

Obviously, $y_{j^*}^* \ne \boldsymbol{0}$. Let $j_\downarrow \triangleq \max\{j = 0, 1, \cdots, j^* : \, y_j^* = \boldsymbol{0}\}$ with the convention that $y_0^* = \boldsymbol{0}, y_{0,n}^* = x_n$. We must have $j_\downarrow < j^*$ and

- $\lim_n y_{j_\downarrow,n}^* = y_{j_\downarrow}^* = \boldsymbol{0}$

- For any $j = j_\downarrow + 1, j_\downarrow + 2, \cdots, j^*, \lim_n t_{j,n} = t_j^* < \infty$.

Now using the continuity of the ODE flow, we must have that

$$\lim_n \widetilde{\boldsymbol{x}}\Big( \sum_{j=j_\downarrow+1}^{j^*} t_{j,n}, y_{j_\downarrow,n}^*; \, (t_{j_\downarrow+1,n}, t_{j_\downarrow+2,n}, \cdots, t_{j^*,n}), (w_{j_\downarrow+1,n}, w_{j_\downarrow+2,n}, \cdots, w_{j^*,n}) \Big)$$

$$= \widetilde{\boldsymbol{x}}\Big( \sum_{j=j_\downarrow+1}^{j^*} t_j^*, \boldsymbol{0}; \, (t_{j_\downarrow+1}^*, t_{j_\downarrow+2}^*, \cdots, t_{j^*}^*), (w_{j_\downarrow+1}^*, w_{j_\downarrow+2}^*, \cdots, w_{j^*}^*) \Big) = y_{j^*}^*.$$

However, due to $\boldsymbol{d}(y_{j^*}^*, \mathcal{G}^c) \le \epsilon_0$ and recall our choice of $\epsilon_0 = \epsilon_1/2$, for all $n$ sufficiently large, we must have

$$\boldsymbol{d}\Big( \widetilde{\boldsymbol{x}}\Big( \sum_{j=j_\downarrow+1}^{j^*} t_{j,n}, y_{j_\downarrow,n}^*; \, (t_{j_\downarrow+1,n}, t_{j_\downarrow+2,n}, \cdots, t_{j^*,n}), (w_{j_\downarrow+1,n}, w_{j_\downarrow+2,n}, \cdots, w_{j^*,n}) \Big), \mathcal{G}^c \Big) \le \frac{5}{8}\epsilon_1.$$

Meanwhile, using Gronwall's inequality repeatedly and $\sup_{j=j_\downarrow+1,\cdots,j^*, n \ge 1} t_{j,n} < \infty$, by substituting the initial condition $y_{j_\downarrow,n}^*$ with $\boldsymbol{0}$, we have

$$\lim_n \Big\| \widetilde{\boldsymbol{x}}\Big( \sum_{j=j_\downarrow+1}^{j^*} t_{j,n}, y_{j_\downarrow,n}^*; \, (t_{j_\downarrow+1,n}, t_{j_\downarrow+2,n}, \cdots, t_{j^*,n}), (w_{j_\downarrow+1,n}, w_{j_\downarrow+2,n}, \cdots, w_{j^*,n}) \Big)$$

$$- \widetilde{\boldsymbol{x}}\Big( \sum_{j=j_\downarrow+1}^{j^*} t_{j,n}, \boldsymbol{0}; \, (t_{j_\downarrow+1,n}, t_{j_\downarrow+2,n}, \cdots, t_{j^*,n}), (w_{j_\downarrow+1,n}, w_{j_\downarrow+2,n}, \cdots, w_{j^*,n}) \Big) \Big\| = 0.$$

This implies that, for all $n$ sufficiently large,

$$\boldsymbol{d}\Big( \widetilde{\boldsymbol{x}}\Big( \sum_{j=j_\downarrow+1}^{j^*} t_{j,n}, \boldsymbol{0}; \, (t_{j_\downarrow+1,n}, t_{j_\downarrow+2,n}, \cdots, t_{j^*,n}), (w_{j_\downarrow+1,n}, w_{j_\downarrow+2,n}, \cdots, w_{j^*,n}) \Big), \mathcal{G}^c \Big) \le \frac{3}{4}\epsilon_1.$$

However, this contradicts eq. (K.7). This implies the existence of the required $\delta_0$ and concludes the proof. $\qquad\square$

### K.1 SUFFICIENT CONDITIONS FOR ASSUMPTION I.9

In this section, we show that under a proper set of regularity conditions on the boundary set $\partial\mathcal{G}$ and the distribution of noises $Z_n$, Assumption I.9 will hold for (Lebesgue) almost every $b > 0$. In particular, we stress that the $C^2$ condition about manifold $\partial\mathcal{G}$ in Assumption I.3, as well as the condition that measures $S_j$ are absolutely continuous w.r.t. the spherical measure $\boldsymbol{\sigma}$ on $\mathbb{S}^{d-1}$, are only used to prove that $\mu\big(h^{-1}(\partial\mathcal{G})\big) = 0$ and will only be applied in this section.

The key of our argument is the following geometric observation regarding the intersection of $C^2$−manifold $\partial\mathcal{G}$ and $\partial B$ of some ball $B$. We stress that, as made evident by the proof, this lemma is essentially based on two assumptions: (I) As the boundary set of the connected bounded region $\mathcal{G}$, $\partial\mathcal{G}$ is a closed set in $\mathbb{R}^d$; (II) As a $(d-1)$-dimensional manifold, $\partial\mathcal{G}$ is of class $C^2$.

**Lemma K.1.** *Let $\boldsymbol{\sigma}_{x,b}$ be the spherical measure on the sphere of the open ball $B(x,b)$ for any $x \in \mathbb{R}^d, b > 0$. Under Assumptions I.1 and I.3, it holds for (Lebesgue) almost every $b > 0$ that*

$$\boldsymbol{\sigma}_{x,b}\big(\partial\mathcal{G} \cap \partial B(x,b)\big) = 0 \quad \forall x \in \mathbb{R}^d. \tag{K.8}$$

*Proof.* The fact that $\partial\mathcal{G}$ is a subset of a separable metric space implies the existence of a countable atlas for this manifold. Therefore, we can find a sequence $(U_i)_{i \geq 1}$ that are bounded open subsets of $\mathbb{R}^{d-1}$ containing $\mathbf{0}$, a sequence $(V_i)_{i \geq 1}$ that are open sets in $\partial\mathcal{G}$ (in the metric space induced by Euclidean distance), and a sequence of injective $C^2$ mapping $f_i$ with $f_i : \mathbb{R}^{d-1} \mapsto \mathbb{R}^d$ and $f_i(U_i) = V_i$ such that $\partial\mathcal{G} = \cup_i V_i$.

Next, we zoom in on a specific chart $(U_i, V_i, f_i)$ and observe the following facts. For any $x \in U_i, y \in V_i$ with $f_i(x) = y$, there exist some orthogonal matrix $Q_y \in \mathbb{R}^{d \times d}$ such that $Q_y n(y) = (0,0,\cdots,0,1)^T$ where the vector field $n(\cdot)$ is the outer normal on $\partial\mathcal{G}$. Besides, there is some vector $a_y \in \mathbb{R}^d$ such that $Q_y y + a_y = \mathbf{0}$. Moreover, there exist an open set on $\widetilde{U}_x$ in $\mathbb{R}^{d-1}$ containing $\mathbf{0}$, an open set $y \in V_y \subset V_i$, an open set $\widetilde{V}_y \triangleq \{Q_y w + a_y : w \in V_y\}$, a $C^2$ function $\widetilde{g}_y : \widetilde{U}_x \mapsto \mathbb{R}$ satisfying $\widetilde{g}_y(\mathbf{0}) = 0$ and

$$\widetilde{g}_y(w_1,\cdots,w_{d-1}) = w_d \quad \forall \boldsymbol{w} = (w_1,\cdots,w_d) \in \widetilde{V}_y.$$

In other words, for any given $y$ on this chart we simply rotate the chart to ensure that the tangent space at $y$ after rotation is $\{(x_1,\cdots,x_{d-1},0) : x_i \in \mathbb{R} \ \forall i \in [d-1]\}$, and reparametrize the $C^2$ diffeomorphism associated to this chart around $y$ so that the coordinates are simply the projection onto the said tangent space. In this sense, the (rotated) manifold is also the graph of the $C^2$ mapping $g_y$. This allows us to define (for any $y \in \partial\mathcal{G}$)

$$A(y) \triangleq \Big(\lambda_1\big(\nabla^2 g_y(\mathbf{0})\big), \lambda_2\big(\nabla^2 g_y(\mathbf{0})\big), \cdots, \lambda_{d-1}\big(\nabla^2 g_y(\mathbf{0})\big)\Big)$$

where, for any real symmetric $(d-1) \times (d-1)$ matrix $A$, $\lambda_1(A) \geq \lambda_2(A) \geq \cdots \geq \lambda_{d-1}(A)$ are the ordered eigenvalues of the matrix, and $\nabla^2 g_y(\cdot)$ is the Hessian of $g_y$. It is worth noticing that $A(\cdot)$ is a continuous function (on $\partial\mathcal{G}$) due to the manifold being of class $C^2$.

Restricting our discussion on some fixed chart $(U_i, V_i, f_i)$ for now, for any $b > 0$, let

$$\mathcal{A}_i(b) \triangleq \{x \in U_i : \text{ for } y = f_i(x), \exists j \in [d-1] \text{ such that } |\lambda_j\big(\nabla^2 g_y(\mathbf{0})\big)| = 1/b\}.$$

Note that for any $b > 0$, the set $\mathcal{A}_i(b)$ is a closed set (hence Borel measurable) since the continuity of $A(\cdot)$ implies that $(\mathcal{A}_i(b))^c$ is an open set on $U_i$. Furthermore, since $\boldsymbol{m}_{\text{Leb}}(U_i) < \infty$, there are at most countably many $b > 0$ such that $\boldsymbol{m}_{\text{Leb}}(\mathcal{A}_i(b)) > 0$. Given the countability of the atlas, we know that

$$\mathcal{B}^* \triangleq \{b > 0 : \exists i \in \mathbb{N} \text{ s.t. } \boldsymbol{m}_{\text{Leb}}(\mathcal{A}_i(b)) > 0\}$$

contains at most countably many elements. In the rest of this proof, we show that eq. (K.8) holds for any $b > 0$ such that $b \notin \mathcal{B}^*$.

Henceforth, we arbitrarily choose some $b > 0$ such that $b \notin \mathcal{B}^*$. We also arbitrarily choose some $x \in \mathbb{R}^d$ and let $\mathbf{B} = B(x,b)$. To facilitate the discussion, we introduce a concept that is closely related to the set $\mathcal{A}_i(b)$: let

$$\mathcal{A}_i^{\mathcal{G}}(b) \triangleq \{y \in V_i : f_i^{-1}(y) \in \mathcal{A}_i(b)\} = \{y \in V_i : A(y) = (b,b,\cdots,b) \text{ or } (-b,-b,\cdots,-b)\}$$

and let $\mathcal{A}^{\mathcal{G}}(b) \triangleq \cup_{i \geq 1} \mathcal{A}_i^{\mathcal{G}}(b)$. Now consider the following decomposition of the sphere $\partial \mathbf{B}$:

$$\mathbf{B}_2 \triangleq \partial \mathbf{B} \cap \mathcal{A}^{\mathcal{G}}(b),$$
$$\mathbf{B}_1 \triangleq \partial \mathbf{B} \backslash \mathbf{B}_2.$$

First, note that for any $y \in \mathbf{B}_1$, one of the following four cases has to occur:

- $y \notin \partial \mathcal{G}$;

- $y \in \partial \mathcal{G}$ and the vector $y - x$ lies in $T_y \partial \mathcal{G}$, the tangent space of the manifold $\partial \mathcal{G}$ at $y$;

- $y \in \partial \mathcal{G}$; the vector $y - x$ is not in $T_y \partial \mathcal{G}$ yet it is not orthogonal to $T_y \partial \mathcal{G}$ either, i.e. $y - x$ is not equal to $c \cdot n(y)$ for any $c \in \mathbb{R}$ where $n(y)$ is the outer normal at $y$;

- $y \in \partial \mathcal{G}$; the vectors $y - x$ and $n(y)$ are linearly dependent.

Our next goal is to show that, in any of these four cases, we can always find a set $y \in O_y$ that is open on the sphere $\partial \mathbf{B}$ such that $\boldsymbol{\sigma}_{x,b}(O_y \cap \partial \mathcal{G}) = 0$. Note that this is obviously true when $y \notin \partial \mathcal{G}$, since both $\partial \mathcal{G}$ and $\partial \mathbf{B}$ are closed sets in $\mathbb{R}^d$. Now we consider the second case. If the vector $y - x$ lies in $T_y \partial \mathcal{G}$, then after applying the affine transformation with orthogonal matrix $Q_y$ (recall that $Q_y n(y) = (0, \cdots, 0, 1)$), we have

$$Q_y(y - x) \in Q_y T_y \partial \mathcal{G} = \{(w_1, \cdots, w_{d-1}, 0) : w_i \in \mathbb{R} \, \forall i \in [d-1]\}.$$

Since $Q_y y + a_y = \mathbf{0}$, we now know that after the affine transformation, the center of the ball $\mathbf{B}$ moves to

$$Q_y x + a_y = Q_y y + a_y + Q_y(x - y) \in \{(w_1, \cdots, w_{d-1}, 0) : w_i \in \mathbb{R} \, \forall i \in [d-1]\}.$$

Without loss of generality, we can assume that $Q_y x + a_y = (b, 0, \cdots, 0, 0)$. In other words, after the affine transformation, the ball becomes $\widetilde{\mathbf{B}}_y \triangleq Q_y \mathbf{B} + a_y = B\big((b, 0, 0, \cdots, 0), b\big)$. Moreover, for any $\boldsymbol{w} = (w_1, \cdots, w_d) \in \partial \widetilde{\mathbf{B}}_y$, we must have $w_1 \geq 0$ and

$$(w_1 - b)^2 + (w_2)^2 + \cdots + (w_d)^2 = b^2.$$

Meanwhile, from the definition of the mapping $\widetilde{g}_y$, one can see that there is an open set $U_y$ around $y$ such that for any $w \in U_y$ with $w \in \partial \mathcal{G} \cap \partial \mathbf{B}$ (hence $\widetilde{w} = Q_y w + a_y \in \widetilde{V}_y \in \partial \widetilde{B}_y$) such that

$$\widetilde{g}_y(\widetilde{w})^2 = b^2 - (\widetilde{w}_1 - b)^2 - \widetilde{w}_2^2 - \cdots - \widetilde{w}_{d-1}^2.$$

Let $\widetilde{f}(w_1, \cdots, w_{d-1}) \triangleq \widetilde{g}_y^2(w_1, \cdots, w_{d-1}) + w_1^2 - 2bw_1 + w_2^2 + \cdots + w_{d-1}^2$. Now we now that $\widetilde{f}(\mathbf{0}) = 0$ and $\widetilde{f}(\widetilde{w}) = 0$ for any $\widetilde{w} \in \widetilde{V}_y \in \partial \widetilde{B}_y$. Moreover, by definition we have $\nabla \widetilde{g}_y(\mathbf{0}) = \mathbf{0}$ so $\frac{\partial}{\partial w_1} \widetilde{f}(\mathbf{0}) = -2b \neq 0$. Due to implicit function theorem, we now know the existence of some open set $U^*$ in $\mathbb{R}^d$ containing $\mathbf{0}$, some $C^2$ function $g^* : \mathbb{R}^{d-2} \mapsto \mathbb{R}$ such that for any $w \in \widetilde{V}_y \cap \partial \widetilde{\mathbf{B}}_y \cap U^*$,

$$w_1 = g^*(w_2, \cdots, w_{d-1}), \quad w_d = \widetilde{g}_y(w_1, \cdots, w_{d-1}).$$

Therefore, within some open neighborhood $V_y^*$ of such $y$, the set $V_y^* \cap \partial \mathbf{B} \cap \partial \mathcal{G}$ is a submanifold with dimension $d - 2$, so we must have

$$\boldsymbol{\sigma}_{x,b}(V_y^* \cap \partial \mathbf{B} \cap \partial \mathcal{G}) = 0.$$

Next, we consider the case where $y \in \partial \mathcal{G} \cap \mathbf{B}_1$ but $y - x$ is neither in $T_y \partial \mathcal{G}$ nor orthogonal to $T_y \partial \mathcal{G}$. Similar to the construction of the affine transformation with $Q_y, a_y$ above, one can find an orthogonal matrix $\widetilde{Q}_y$ and a vector $\widetilde{a}_y$ such that $\widetilde{Q}_y y + a_y = \mathbf{0}$ and $\widetilde{Q}_y(y - x) = (0, 0, \cdots, 0, b)$. Let $\widetilde{\mathbf{B}} = B((0, 0, \cdots, -b), b)$. In other words, this time the rotation we constructed ensures that, after rotation, the vector between $\widetilde{Q}_y y + a_y = \mathbf{0}$ (on the sphere $\partial \widetilde{\mathbf{B}}$) and the center $\widetilde{Q}_y x + \widetilde{a}_y$ of the ball $\widetilde{\mathbf{B}}$ is aligned with the $d$-th axis. Moreover, there is an open set $y \in V_y^{\mathrm{alt}} \subset \mathbb{R}^d$ and a $C^2$ function $\widetilde{g}_y^{\mathrm{alt}} : \mathbb{R}^{d-1} \mapsto \mathbb{R}$ with $\nabla \widetilde{g}_y^{\mathrm{alt}}(\mathbf{0}) \neq \mathbf{0}$ such that for any $\widetilde{y} \in \widetilde{V}_y^{\mathrm{alt}} \triangleq \widetilde{Q}_y V_y^{\mathrm{alt}} + \widetilde{a}_y$, we have

$$\widetilde{y}_d = \widetilde{g}_y^{\mathrm{alt}}(\widetilde{y}_1, \cdots, \widetilde{y}_{d-1}).$$

By applying a change of coordinates if necessary (which can be achieved by multiplying another orthogonal matrix), we can assume without loss of generality that $\nabla \widetilde{g}_y^{\mathrm{alt}}(\mathbf{0}) = (0, 0, \cdots, c)$ for some $c \neq 0$. Then for any $\widetilde{w} \in V_y^{\mathrm{alt}}$ such that $w \in \partial \mathbf{B} \cap \partial \mathcal{G}$, let $w = \widetilde{Q}_y \widetilde{w} + \widetilde{a}_y$ and note that we must have

$$w_1^2 + w_2^2 + \cdots + w_{d-1}^2 + \big(b + \widetilde{g}_y^{\mathrm{alt}}(w_1, \cdots, w_{d-1})\big)^2 = b^2.$$

In particular, $\widetilde{g}_y^{\mathrm{alt}}(w_1, \cdots, w_{d-1}) = cw_{d-1} + r(w)$ for some $C^2$ function $r$ with $r(\mathbf{0}) = 0, \nabla r(\mathbf{0}) = \mathbf{0}$. As a result, for function $\widetilde{f}(w) \triangleq w_1^2 + w_2^2 + \cdots + w_{d-1}^2 + \big(b + \widetilde{g}_y^{\mathrm{alt}}(w_1, \cdots, w_{d-1})\big)^2$. we have $\frac{\partial}{\partial w_{d-1}} \widetilde{f}(\mathbf{0}) = 2bc \neq 0$. Using implicit function theorem again, one can see the existence of some open set $\mathbf{0} \in V_y^* \in \mathbb{R}^d$ and some $C^2$ function $g^* : \mathbb{R}^{d-2} \mapsto \mathbb{R}$ such that for any $\widetilde{w} \in V_y^* \cap \partial \mathcal{G} \cap \partial \mathbf{B}$, we have that (for $w \triangleq \widetilde{Q}_y \widetilde{w} + \widetilde{a}_y$)

$$w_{d-1} = g^*(w_1, \cdots, w_{d-2}), \quad w_d = \widetilde{g}_y^{\mathrm{alt}}(w_1, \cdots, w_{d-1}).$$

Again, we have established that within some open neighborhood $V_y^*$ of such $y$, the set $V_y^* \cap \partial \mathbf{B} \cap \partial \mathcal{G}$ is a submanifold with dimension $d - 2$, so we must have

$$\boldsymbol{\sigma}_{x,b}(V_y^* \cap \partial \mathbf{B} \cap \partial \mathcal{G}) = 0.$$

Lastly, consider the case where $y \in \partial \mathcal{G} \cap \mathbf{B}_2$ and the vectors $y - x$ and $n(y)$ are linearly dependent. In other words, the tangent space $T_y \partial \mathcal{G}$ is also the tangent space $T_y \partial \mathbf{B}$. Since $y \notin \mathbf{B}_1$, we know that $\widetilde{g}_y(w) = \frac{1}{2} w^T A_y w + r_1(w)$ where $A_y$ is a real symmetric matrix with no eigenvalue equal to $\pm 1/b$ and $r_1$ is a $C^2$ function with $|r_1(w)| = o(\|w\|^2)$. On the other hand, for any $\widetilde{w}$ in the open set $V_y \subset \partial \mathcal{G}$, if we also have $\widetilde{w} \in \partial \mathbf{B} \cap \partial \mathcal{G}$, then for $w \triangleq Q_y \widetilde{w} + a_y$ we have

$$w_d = \widetilde{g}_y(w_1, \cdots, w_{d-1}) = \sqrt{b^2 - w_1^2 - w_2^2 - \cdots - w_{d-1}^2} - b. \tag{K.9}$$

Also, note that $\sqrt{b^2 - w_1^2 - w_2^2 - \cdots - w_{d-1}^2} - b = -\frac{1}{2}(w_1, \cdots, w_{d-1}) \frac{\mathbf{I}_{d-1}}{b} (w_1, \cdots, w_{d-1})^T + r_2(w_1, \cdots, w_{d-1})$ where $r_2$ is also a $C^2$ function with $|r_2(w)| = o(\|w\|^2)$. Therefore, for any $(w_1, \cdots, w_{d-1})$ satisfying the equation eq. (K.9), we have

$$\frac{1}{2}(w_1, \cdots, w_{d-1})\big(A_y - \frac{\mathbf{I}_{d-1}}{b}\big)(w_1, \cdots, w_{d-1})^T = -r_1(w_1, \cdots, w_{d-1}) + r_2(w_1, \cdots, w_{d-1}).$$

However, for the real symmetric matrix $A_y - \frac{\mathbf{I}_{d-1}}{b}$, note that none of its eigenvalue is equal to $0$, implying the existence of some $\epsilon > 0$ such that

$$\left| \frac{1}{2}(w_1, \cdots, w_{d-1})\big(A_y - \frac{\mathbf{I}_{d-1}}{b}\big)(w_1, \cdots, w_{d-1})^T \right| \geq \epsilon(w_1^2 + \cdots + w_{d-1}^2).$$

For this fixed $\epsilon > 0$, we can also find $\delta > 0$ such that

$$|-r_1(w_1, \cdots, w_{d-1}) + r_2(w_1, \cdots, w_{d-1})| \leq \frac{\epsilon}{2}(w_1^2 + \cdots + w_{d-1}^2)$$

for any $w_1^2 + \cdots + w_{d-1}^2 < \delta$. As a result, the only solution to eq. (K.9) with $w_1^2 + \cdots + w_{d-1}^2 < \delta$ is $w_1 = w_2 = \cdots, w_{d-1} = 0$. In summary, we have shown that there exists some set $y \in V_y^*$ that is open in $\mathbb{R}^d$ such that $V_y^* \cap \partial \mathbf{B} \cap \partial \mathcal{G} = \{y\}$.

Collecting the results we have established so far, we now know that for any $y \in \mathbf{B}_1$ (recall that $\mathbf{B}_1$ is an open set on $\partial \mathbf{B}$), there is an open set $V_y^*$ containing $y$ and satisfying $\boldsymbol{\sigma}_{x,b}(V_y^* \cap \partial \mathbf{B} \cap \partial \mathcal{G}) = 0$. In particular, given the open cover $\cup_{y \in \mathbf{B}_2} V_y^* = \mathbf{B}_1$, Lindelöf property then allows us to extract a countable open $\cup_{i \geq 1} V_{y_i}^* = \mathbf{B}_1$ cover and conclude that

$$\boldsymbol{\sigma}_{x,b}(\mathbf{B}_1 \cap \partial \mathcal{G}) \leq \sum_{i \geq 1} \boldsymbol{\sigma}_{x,b}(V_{y_i}^* \cap \partial \mathbf{B} \cap \partial \mathcal{G}) = 0.$$

Moving on, we evaluate $\boldsymbol{\sigma}_{x,b}(\mathbf{B}_2 \cap \partial \mathcal{G})$. For any $y \in \partial \mathbf{B}$, one of the four cases has to occur:

- $y \notin \mathcal{A}^{\mathcal{G}}(b)$;

- $y \in \mathcal{A}^{\mathcal{G}}(b)$ and the vector $y - x$ lies in $T_y \partial \mathcal{G}$, the tangent space of the manifold $\partial \mathcal{G}$ at $y$;

- $y \in \mathcal{A}^{\mathcal{G}}(b)$; the vector $y - x$ is not in $T_y \partial \mathcal{G}$ yet it is not orthogonal to $T_y \partial \mathcal{G}$ either, i.e. $y - x$ is not equal to $c \cdot n(y)$ for any $c \in \mathbb{R}$ where $n(y)$ is the outer normal at $y$;

- $y \in \mathcal{A}^{\mathcal{G}}(b)$; the vectors $y - x$ and $n(y)$ are linearly dependent.

Again, we show that in any of these four cases, there is some set $y \in V_y^*$ open in $\partial \mathbf{B}$ such that $\boldsymbol{\sigma}_{x,b}(\mathcal{A}^{\mathcal{G}}(b) \cap V_y^*) = 0$. In the first case, the fact that $\mathcal{A}^{\mathcal{G}}(b)$ is closed on $\partial \mathcal{G}$ immediately implies the existence of some $V_y^*$ such that $\mathcal{A}^{\mathcal{G}}(b) \cap V_y^* = \emptyset$. For the second and third case, this can be shown using exactly the same implicit function argument above. For the last case where $y \in \mathcal{A}^{\mathcal{G}}(b) \cap \partial \mathbf{B}$ and the vectors $y - x$ and $n(y)$ are linearly dependent, from $y \in \mathcal{A}^{\mathcal{G}}(b) \cap \partial \mathbf{B}$ we know that $y \in V_i$ where $(U_i, V_i, f_i)$ is a chart of $\partial \mathcal{G}$ and $V_i$ is open on $\partial \mathcal{G}$. Moreover, recall the construction of open set $y \in V_y \subset V_i$ at the beginning of the proof. It is worth noticing that $V_i \cap \mathcal{A}^{\mathcal{G}}(b) = V_i \cap \mathcal{A}_i^{\mathcal{G}}(b)$. Besides, due to the fact that the vectors $y - x$ and $n(y)$ are linearly dependent, we know that the tangent space $T_y \partial \mathcal{G}$ is also the tangent space $T_y \partial \mathbf{B}$. Therefore, by definition of $\widetilde{g}_y$ and $Q_y, a_y$, we know that $Q_y y + a_y = \mathbf{0}$, and under the affine transformation, the ball becomes $B((0, 0, \cdots, \pm b), b)$. Without loss of generality, we assume it is $\widetilde{\mathbf{B}} \triangleq B((0, 0, \cdots, b), b)$. Moreover, since $b \notin \mathcal{B}^*$, we have that $\boldsymbol{m}_{\mathrm{Leb}}(\{w \in \widetilde{U}_y : x \in \mathcal{A}_i(b)\}) = 0$ where, as defined at the beginning of the proof, $\widetilde{U}_y$ is the domain of the $C^2$ mapping $\widetilde{g}_y$, $V_y$ is the image of the mapping, and $\widetilde{V}_y \triangleq \{Q_y w + a_y : w \in V_y\}$ is the image of $V_y$ under the affine transformation. Therefore, for any $\widetilde{w} \in \mathcal{A}_i^{\mathcal{G}}(b) \cap \partial \mathbf{B}$, let $w = Q_y \widetilde{w} + a_y$ and we must have

$$w_d = \widetilde{g}_y(w_1, \cdots, w_{d-1}) = -b + \sqrt{b^2 - w_1^2 - w_2^2 - \cdots - w_{d-1}^2},$$

$$(w_1, \cdots, w_{d-1}) \in \{w \in \widetilde{U}_y : x \in \mathcal{A}_i(b)\}.$$

Now let $V_y^* \triangleq \Big\{ Q_y^T(v - a_y) : v = (w_1, \cdots, w_{d-1}, -b + \sqrt{b^2 - w_1^2 - w_2^2 - \cdots - w_{d-1}^2}) \text{ for some } w \in \widetilde{U}_x \Big\}$ and note that $y \in V_y^*$ is an open set on $\partial \mathbf{B}$. (Specifically, note that we simply identify an open set on the transformed sphere $\partial \widetilde{B}_y$, and then perform the inverse transformation to move the set back to the original sphere $\partial \mathbf{B}$.) Then it follows immediately from $\boldsymbol{m}_{\mathrm{Leb}}(\{w \in \widetilde{U}_y : x \in \mathcal{A}_i(b)\}) = 0$ that $\boldsymbol{\sigma}_{x,b}(\mathcal{A}^{\mathcal{G}}(b) \cap V_y^*) = 0$.

In summary, for any $y \in \partial \mathbf{B}$, we can find a set $y \in V_y^*$ open on $\partial \mathbf{B}$ such that $\boldsymbol{\sigma}_{x,b}(\mathcal{A}^{\mathcal{G}}(b) \cap V_y^*) = 0$. Lastly, by applying Lindelöf property again, we extract a countable open cover $\cup_{i \geq 1} V_{y_i}^* = \mathbf{B}_2$ cover and conclude that

$$\boldsymbol{\sigma}_{x,b}(\partial \mathbf{B} \cap \mathcal{A}^{\mathcal{G}}(b)) \leq \sum_{i \geq 1} \boldsymbol{\sigma}_{x,b}(\mathcal{A}^{\mathcal{G}}(b) \cap V_{y_i}^*) = 0$$

and this concludes the proof. $\qquad\square$

As a result of Lemma K.1, the following Lemma is essentially built upon three assumptions/facts: (I) As a boundary set, $\partial \mathcal{G}$ is a closed set in $\mathbb{R}^d$; (II) As a $(d-1)$-dimensional manifold, $\partial \mathcal{G}$ is of class $C^2$; (III) The measures $S_j$ in Assumption I.6 are absolutely continuous w.r.t. the spherical measure $\boldsymbol{\sigma}$.

**Lemma K.2.** *Under Assumptions I.1, I.3 and I.6, it holds for (Lebesgue) almost every $b > 0$ that*

$$\mu\Big(h^{-1}(\partial \mathcal{G})\Big) = 0.$$

*Proof.* Fix some $b > 0$ satisfying the conditions in Lemma K.1. Recall that $\mu = \sum_{j \in j(i^*)} \mu_j$ (see eq. (J.40)). It suffices to show that $\mu_j\Big(h^{-1}(\partial \mathcal{G})\Big) = 0$ for some fixed $j \in j(i^*)$. In particular,

observe that

$$
\mu_{\boldsymbol{j}}\Big(h^{-1}\big(\partial\mathcal{G}\big)\Big)
$$
$$
= \int_{t_{i+1}>0,\ \theta_i\in\mathbb{S}^{d-1},\ r_i>0\ \forall i\in[k^*-1]}
$$
$$
\cdot\Big(\int_{\theta_{k^*}\in\mathbb{S}^{d-1},r_{k^*}>0}\mathbb{1}\big\{h^*(r_1,\cdots,r_{k^*-1},\theta_1,\cdots,\theta_{k^*-1},t_2,\cdots,t_{k^*})+\varphi_b(r_{k^*}\theta_{k^*})\in\partial\mathcal{G}\big\}
$$
$$
\cdot S_{\boldsymbol{j}_{k^*}}(d\theta_{k^*})\nu_{\alpha_{j_{k^*}}}(dr_{k^*})\Big)\cdot\prod_{i=1}^{k^*-1}\nu_{\alpha_{j_i}}(dr_i)\times S_{\boldsymbol{j}_i}(d\theta_i)\times\boldsymbol{m}_{\mathrm{Leb}}(dt_{i+1}) \tag{K.10}
$$

where the function $h^*$ is defined as

$$
h^*(r_1,\cdots,r_{k^*-1},\theta_1,\cdots,\theta_{k^*-1},t_2,\cdots,t_{k^*})
$$
$$
\triangleq\widetilde{\boldsymbol{x}}\Big(\sum_{i=2}^{k^*-1}t_i,\boldsymbol{0};(0,t_2,\cdots,t_{k^*-1}),(r_1\theta_1,r_2\theta_2,\cdots,r_{k^*-1}\theta_{k^*-1})\Big)
$$

Let $z\triangleq h^*(r_1,\cdots,r_{k^*-1},\theta_1,\cdots,\theta_{k^*-1},t_2,\cdots,t_{k^*})$. Now by separating the two cases based on whether the truncation operator takes effect or not, we have

$$
\int_{\theta_{k^*}\in\mathbb{S}^{d-1},r_{k^*}>0}\mathbb{1}\big\{h^*(r_1,\cdots,r_{k^*-1},\theta_1,\cdots,\theta_{k^*-1},t_2,\cdots,t_{k^*})+\varphi_b(r_{k^*}\theta_{k^*})\in\partial\mathcal{G}\big\}
$$
$$
\cdot S_{\boldsymbol{j}_{k^*}}(d\theta_{k^*})\nu_{\alpha_{j_{k^*}}}(dr_{k^*})
$$
$$
= \int_{\theta_{k^*}\in\mathbb{S}^{d-1},r_{k^*}\in(0,b)}\mathbb{1}\big\{z+r_{k^*}\theta_{k^*}\in\partial\mathcal{G}\big\}S_{\boldsymbol{j}_{k^*}}(d\theta_{k^*})\nu_{\alpha_{j_{k^*}}}(dr_{k^*}) \tag{K.11}
$$
$$
+ \int_{\theta_{k^*}\in\mathbb{S}^{d-1}}\mathbb{1}\big\{z+b\theta_{k^*}\in\partial\mathcal{G}\big\}S_{\boldsymbol{j}_{k^*}}(d\theta_{k^*})\cdot\int_{r_{k^*}>b}\nu_{\alpha_{j_{k^*}}}(dr_{k^*}). \tag{K.12}
$$

For term eq. (K.11), note that it is equal to $\int\mathbb{1}\big\{\mathbf{T}^{-1}(r_{k^*},\theta_{k^*})\in(-z+\partial\mathcal{G})\cap\partial B(0,b)\big\}S_{\boldsymbol{j}_{k^*}}(d\theta_{k^*})\times$ $\nu_{\alpha_{j_{k^*}}}(dr_{k^*})$ where $\mathbf{T}^{-1}(r,\theta)=r\theta$ is the inverse of the polar coordinate transform. Furthermore, since the set $(-z+\partial\mathcal{G})\cap\partial B(0,b)$ is either empty or is a $(d-1)$-dimensional $C^2$ submanifold (w.r.t. $B(0,b)$ when viewed as a $d-$dimensional manifold). In other words, it has zero mass under $\boldsymbol{m}_{\mathrm{Leb}}^d$. For the measure $\nu^*\triangleq\mathbf{T}^{-1}\circ(S_{\boldsymbol{j}_{k^*}}\times\nu_{\alpha_{j_{k^*}}})$, due to $S_{\boldsymbol{j}}$ being absolutely continuous w.r.t. $\boldsymbol{\sigma}$, it is easy to see that $\nu^*$ is absolutely continuous w.r.t. $\boldsymbol{m}_{\mathrm{Leb}}^d$. Therefore, we must have

$$
\nu^*\big((-z+\partial\mathcal{G})\cap\partial B(0,b)\big)=0,
$$

implying that the integral in *eq.* (K.11) $=0$. On the other hand, for term eq. (K.12), we know that $\int_{r_{k^*}>b}\nu_{\alpha_{j_{k^*}}}(dr_{k^*})=1/b^{1+\alpha_{j_{k^*}}}<\infty$. Besides,

$$
\int_{\theta_{k^*}\in\mathbb{S}^{d-1}}\mathbb{1}\big\{z+b\theta_{k^*}\in\partial\mathcal{G}\big\}S_{\boldsymbol{j}_{k^*}}(d\theta_{k^*})=\int_{\mathbb{S}^{d-1}}\mathbb{1}\big\{z+b\theta\in\partial\mathcal{G}\cap\partial B(z,b)\big\}S_{\boldsymbol{j}_{k^*}}(d\theta).
$$

Then it follows immediately from Lemma K.1 and $S_{\boldsymbol{j}}$ being absolutely continuous w.r.t. the spherical measure $\boldsymbol{\sigma}$ that the integral in term eq. (K.12) is equal to $0$. In summary, we have shown that

$$
\int_{\theta_{k^*}\in\mathbb{S}^{d-1},r_{k^*}>0}\mathbb{1}\big\{h^*(r_1,\cdots,r_{k^*-1},\theta_1,\cdots,\theta_{k^*-1},t_2,\cdots,t_{k^*})+\varphi_b(r_{k^*}\theta_{k^*})\in\partial\mathcal{G}\big\}
$$
$$
\cdot S_{\boldsymbol{j}_{k^*}}(d\theta_{k^*})\nu_{\alpha_{j_{k^*}}}(dr_{k^*})=0
$$

for any $(r_1,\cdots,r_{k^*-1},\theta_1,\cdots,\theta_{k^*-1},t_2,\cdots,t_{k^*})$. Plug this result back into eq. (K.10) and we conclude the proof. $\qquad\square$

## L  NOTATIONS

Table L.1 lists the notations used in Section G.

Table L.1: Summary of notations frequently used in Section G

| | | |
|---|---|---|
| $[k]$ | $\{1, 2, \ldots, k\}$ | |
| $\eta$ | Learning rate (gradient descent step size) | |
| $b$ | Truncation threshold of stochastic gradient | |
| $\epsilon$ | An accuracy parameter; typically used to denote an $\epsilon-$neighborhood of $s_i, m_i$ | |
| $\delta$ | A threshold parameter used to define *large* noises | |
| $\bar{\epsilon}$ | A constant defined for eq. (G.28)-eq. (G.29). Since $\bar{\epsilon} < \epsilon_0$, in eq. (G.8) the claim holds for $|x - y| < \bar{\epsilon}$. Note that the value of the constant $\bar{\epsilon}$ does not vary with our choice of $\eta, \epsilon, \delta$. | |

| | | |
|---|---|---|
| $M$ | Upper bound of $|f'|$ and $|f''|$ | eq. (G.10) |
| $L$ | Radius of training domain | eq. (G.10) |
| $\Omega$ | The open interval $(s_-, s_+)$; a simplified notation for $\Omega_i$ | |
| $\varphi, \varphi_c$ | $\varphi_c(w) \triangleq \varphi(w, c) \triangleq (w \wedge c) \vee (-c)$ | truncation operator at level $c > 0$ |
| $Z_n^{\leq \delta, \eta}$ | $Z_n \mathbb{1}\{\eta|Z_n| \leq \delta\}$ | "small" noise eq. (G.20) |
| $Z_n^{> \delta, \eta}$ | $Z_n \mathbb{1}\{\eta|Z_n| > \delta\}$ | "large" noise eq. (G.21) |
| $T_j^\eta(\delta)$ | $\min\{n > T_{j-1}^\eta(\delta): \ \eta|Z_n| > \delta\}$ | arrival time of $j$-th large noise eq. (G.22) |
| $W_j^\eta(\delta)$ | $Z_{T_j^\eta(\delta)}$ | size of $j$-th large noise eq. (G.23) |
| $X_n^\eta(x)$ | $X_{n+1}^\eta(x) = \varphi_L\Big(X_n^\eta(x) - \varphi_b\big(\eta(f'(X_n^\eta(x)) - Z_{n+1})\big)\Big), \quad X_0^\eta(x) = x$ | SGD |
| $\mathbf{y}_n^\eta(x)$ | $\mathbf{y}_n^\eta(x) = \mathbf{y}_{n-1}^\eta(x) - \eta f'(\mathbf{y}_{n-1}^\eta(x)), \quad \mathbf{y}_0^\eta(x) = x$ | GD |

| | | |
|---|---|---|
| $Y_n^\eta(x)$ | $\mathbf{y}_n^\eta(x)$ perturbed by large noises $(\mathbf{T}^\eta(\delta), \mathbf{W}^\eta(\delta))$ | GD + large jump |
| $\widetilde{\mathbf{y}}_n^\eta(x; \mathbf{t}, \mathbf{w})$ | $\mathbf{y}_n^\eta(x)$ perturbed by noise vector $(\mathbf{t}, \mathbf{w})$ | perturbed GD |
| $\mathbf{x}^\eta(t, x)$ | $d\mathbf{x}^\eta(t; x) = -\eta f'\Big(\mathbf{x}^\eta(t; x)\Big)dt, \quad \mathbf{x}^\eta(0; x) = x$ | ODE |
| $\mathbf{x}(t, x)$ | $\mathbf{x}^1(t, x)$ | |
| $\widetilde{\mathbf{x}}^\eta(t, x; \mathbf{t}, \mathbf{w})$ | $\mathbf{x}^\eta(t, x)$ perturbed by noise vector $(\mathbf{t}, \mathbf{w})$ | perturbed ODE |
| $A(n, \eta, \epsilon, \delta)$ | $\Big\{ \max\limits_{k \in [n \wedge (T_1^\eta(\delta) - 1)]} \eta|Z_1 + \cdots + Z_k| \leq \epsilon \Big\}.$ | eq. (G.42) |
| $r$ | $r \triangleq \min\{-s_-, s_+\}$. Effective radius of the attraction field $\Omega$. | |
| $l^*$ | $l^* \triangleq \lceil r/b \rceil$. The minimum number of jumps required to escape $\Omega$ when starting from its local minimum $m = 0$. | |
| $h(\mathbf{w}, \mathbf{t})$ | A mapping defined as $h(\mathbf{w}, \mathbf{t}) = \widetilde{\mathbf{x}}(t_{l^*}, 0; \mathbf{t}, \mathbf{w})$. | |
| $\bar{t}, \bar{\delta}$ | Necessary conditions for $h(\mathbf{w}, \mathbf{t})$ to be outside of $\Omega$ | eq. (G.35)-eq. (G.36) |
| $\hat{t}(\epsilon)$ | $\hat{t}(\epsilon) \triangleq c_1 \log(1/\epsilon)$. The quantity $\hat{t}(\epsilon)/\eta$ provides an upper bound for the time it takes $\mathbf{x}^\eta$ to return to $2\epsilon-$neighborhood of local minimum $m = 0$ when starting from somewhere $\epsilon-$away from $s_-, s_+$. See eq. (G.37). | |
| $E(\epsilon)$ | $\big\{ (\mathbf{w}, \mathbf{t}) \subseteq \mathbb{R}^{l^*} \times \mathbb{R}_+^{l^*-1} : h(\mathbf{w}, \mathbf{t}) \notin [(s_- - \epsilon) \vee (-L), (s_+ + \epsilon) \wedge L] \big\}$ | |
| $p(\epsilon, \delta, \eta)$ | The probability that, for $\mathbf{t} = \big(T_j^\eta(\delta) - 1\big)_{j=1}^{l^*}$ and $\mathbf{w} = \big(\eta W_j^\eta(\delta)\big)_{j=1}^{l^*}$, we have $(\mathbf{w}, \mathbf{t}) \in E(\epsilon)$ conditioning on $\{T_1^\eta(\delta) = 1\}$. Intuitively speaking, it characterizes the probability that the first $l^*$ *large* noises alone can drive the ODE out of the attraction field. Defined in eq. (G.66). | |
| $\nu_\alpha$ | The Borel measure on $\mathbb{R}$ with density $$\nu_\alpha(dx) = \mathbb{1}\{x > 0\}\frac{\alpha p_+}{x^{\alpha+1}} + \mathbb{1}\{x < 0\}\frac{\alpha p_-}{|x|^{\alpha+1}}$$ where $p_-, p_+$ are constants in Assumption 2 in the main paper. | |
| $\mu$ | The product measure $\mu = (\nu_\alpha)^{l^*} \times (\mathbf{Leb}_+)^{l^*-1}$. | |
| $\sigma(\eta)$ | $\min\{n \geq 0 : X_n^\eta \notin \Omega\}$. | first exit time |

$H(x)$      $\mathbb{P}(|Z_1| > x) = x^{-\alpha} L(x)$

$T_{\text{return}}(\epsilon, \eta)$      $\min\{n \geq 0 : X_n^{\eta}(x) \in [-2\epsilon, 2\epsilon]\}$

# M RESULTS ABOUT TAIL DISTRIBUTIONS OF NOISES IN OUR NUMERICAL EXPERIMENTS

## M.1 QQ PLOTS

QQ plots below clearly show that the tails in noise distribution are always much lighter than the Pareto distributions with alpha = 2 or even 10. In fact, the tail of noise distributions seem to be between that of lognormal and normal distributions, implying that it is lighter than any power-law distribution.

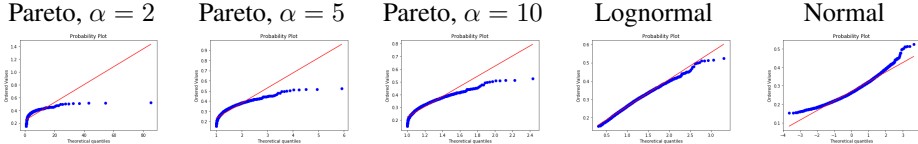

Figure M.1: Ablation Study, Corrupted FMNIST & LeNet: At the beginning

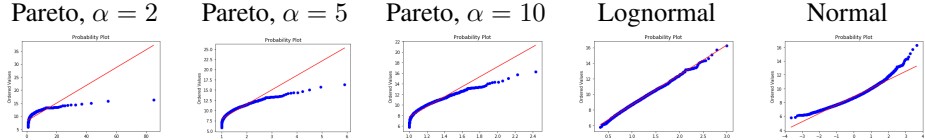

Figure M.2: Ablation Study, Corrupted FMNIST & LeNet: Half way through the training

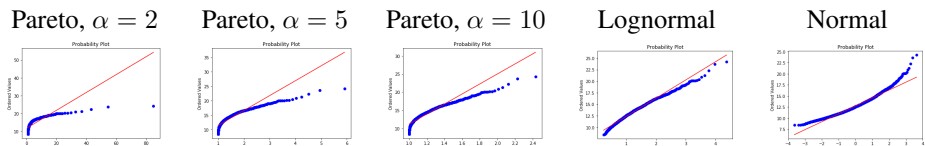

Figure M.3: Ablation Study, Corrupted FMNIST & LeNet: At the end of training

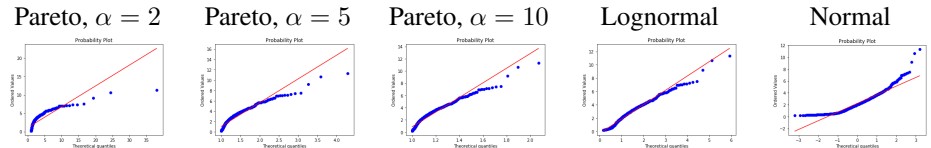

Figure M.4: Ablation Study, SVHN & VGG11: At the beginning

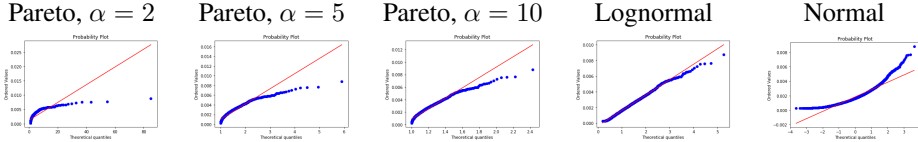

Figure M.5: Ablation Study, SVHN & VGG11: Half way through the training

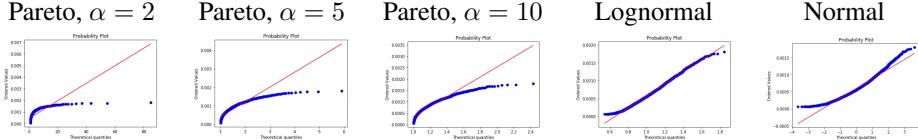

Figure M.6: Ablation Study, SVHN & VGG11: At the end of training

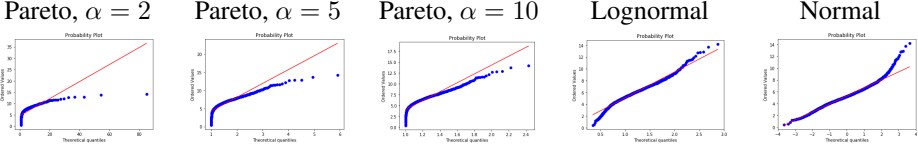

Figure M.7: Ablation Study, CIFAR10 & VGG11: At the beginning

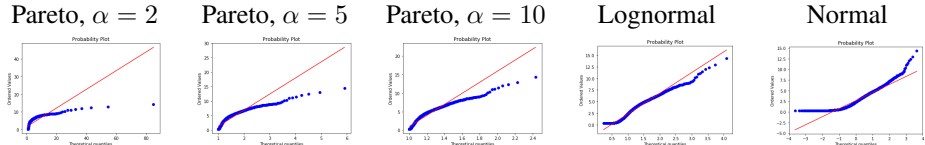

Figure M.8: Ablation Study, CIFAR10 & VGG11: Half way through the training

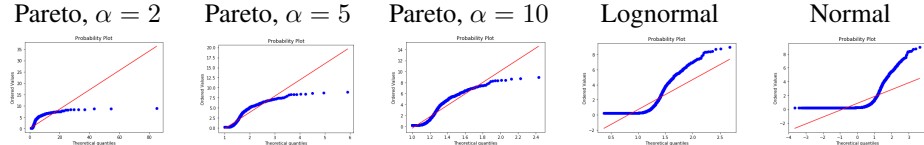

Figure M.9: Ablation Study, CIFAR10 & VGG11: At the end of training

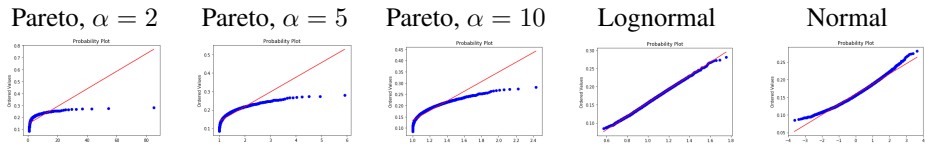

Figure M.10: Data Augmentation, CIFAR10 & VGG11: At the beginning

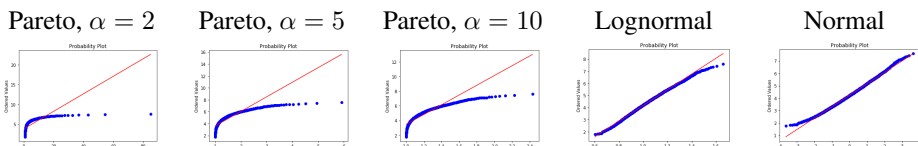

Figure M.11: Data Augmentation, CIFAR10 & VGG11: Half way through the training

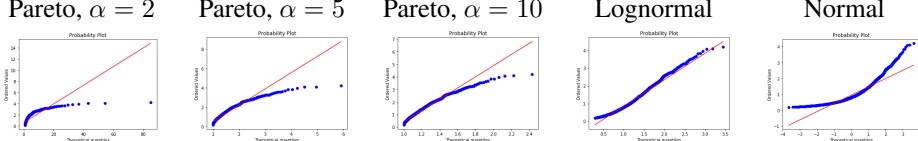

Figure M.12: Data Augmentation, CIFAR10 & VGG11: At the end of training

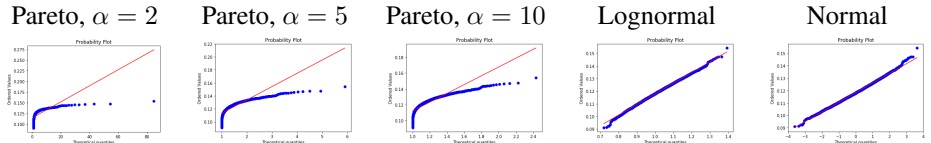

Figure M.13: Data Augmentation, CIFAR100 & VGG16: At the beginning

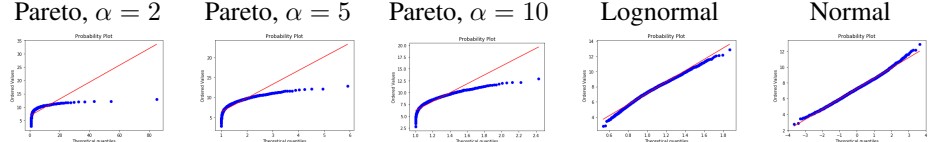

Figure M.14: Data Augmentation, CIFAR100 & VGG16: Halfway through the training

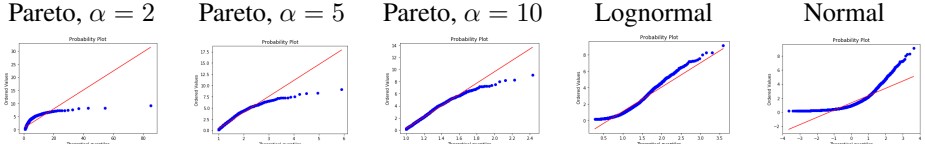

Figure M.15: Data Augmentation, CIFAR100 & VGG16: At the end of training

## M.2 EMPIRICAL MEAN RESIDUAL LIFE (EMRL) PLOTS

It is well known that the mean residual life blows up to infinity if and only if the distribution is heavy-tailed (more precisely, long-tailed). However, from the figures below, one can see that none of the EMRL exhibits such a pattern in any case tested in our experiments. Instead, we see clear downward trends, which strongly suggests light tails in all cases tested.

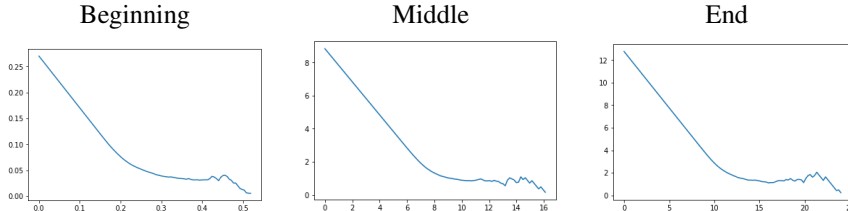

Figure M.16: Plots of empirical mean residual life for noises in FMNIST&LeNet Task throughout training

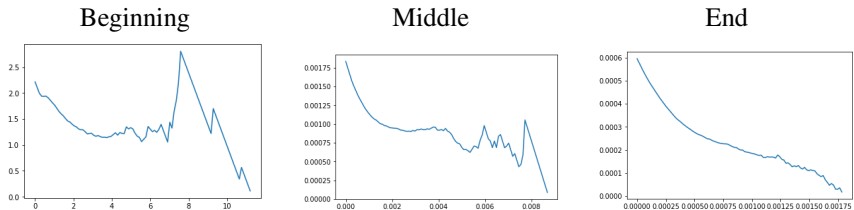

Figure M.17: Plots of empirical mean residual life for noises in SVHN&VGG 11 Task throughout training

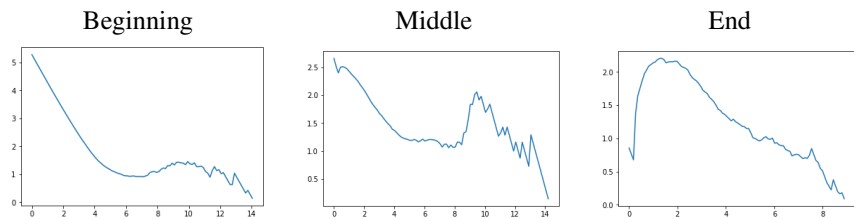

Figure M.18: Plots of empirical mean residual life for noises in CIFAR 10&VGG 11 Task throughout training

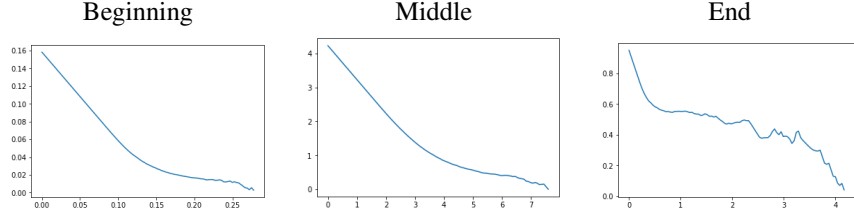

Figure M.19: Plots of empirical mean residual life for noises in dataAug, CIFAR 10&VGG 11 Task throughout training

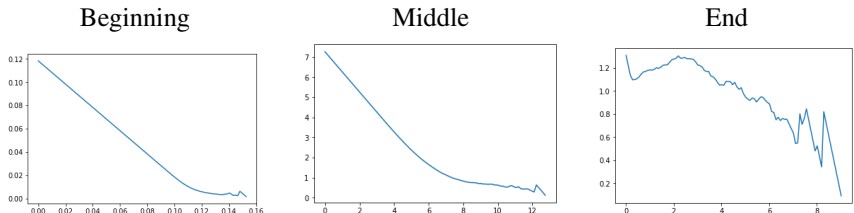

Figure M.20: Plots of empirical mean residual life for noises in dataAug, CIFAR 100&VGG 11 Task throughout training

## M.3  HILL PLOTS

In the hill plots below, the estimated power-law indices using only the top 1% of samples (on the left hand side of the dashed lines) stay well above 10 for the most part and almost never drop below 2. This strongly suggests that even if the gradient noises are from a heavy-tailed distribution, it is likely to have a very high power law index (implying relatively lighter tails), and hence, we cannot expect to observe a prominent heavy-tailed behavior from them.

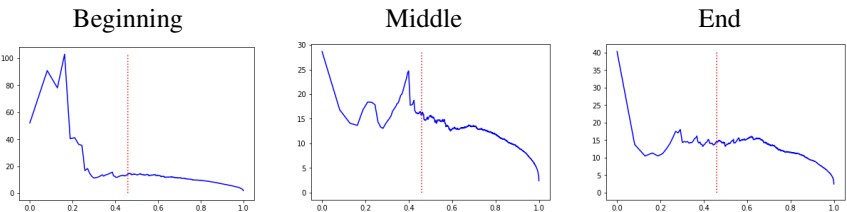

Figure M.21: altHill Plots for noises in FMNIST&LeNet Task throughout training. Dashed Red Line: Estimation based on the largest 1% data

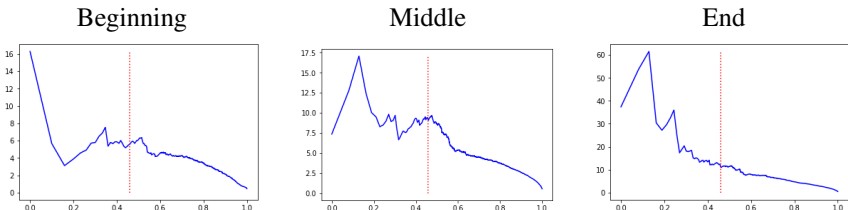

Figure M.22: altHill Plots for noises in SVHN&VGG 11 Task throughout training. Dashed Red Line: Estimation based on the largest 1% data

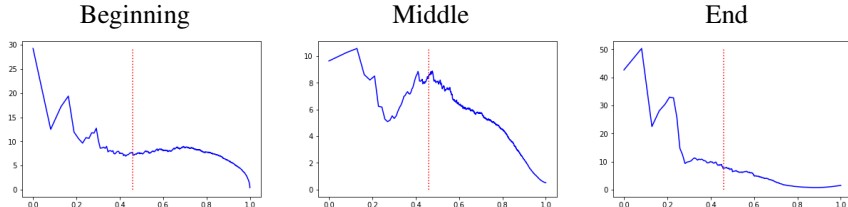

Figure M.23: altHill Plots for noises in CIFAR 10&VGG 11 Task throughout training. Dashed Red Line: Estimation based on the largest 1% data

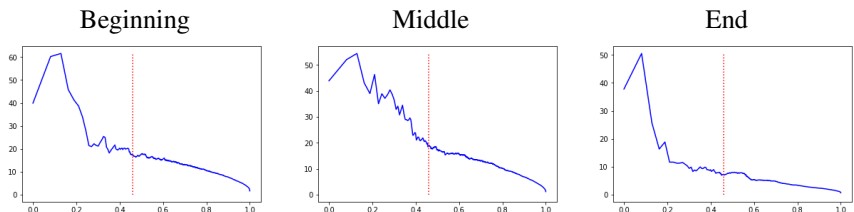

Figure M.24: altHill Plots for noises in Data Augmentation, CIFAR 10&VGG 11 Task throughout training. Dashed Red Line: Estimation based on the largest 1% data

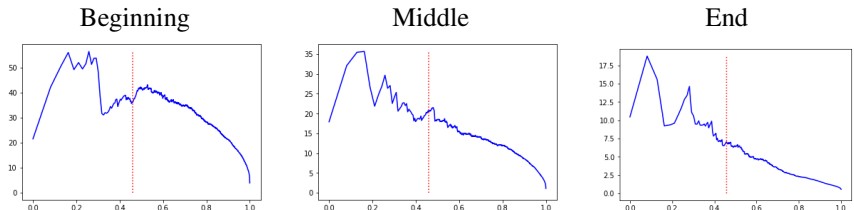

Figure M.25: altHill Plots for noises in Data Augmentation, CIFAR 100&VGG 16 Task throughout training. Dashed Red Line: Estimation based on the largest 1% data

Table M.1: Power-law Indices Estimation throughout the Training, using PLFIT. All the estimations are at least 5 for all cases tested in our experiments, and most of the times the estimation is above 10. This means that even under the assumption that the gradient noises were from a heavy-tailed distribution, they should have much lighter tails than any $\alpha$-stable distribution (which requires $\alpha < 2$) or the heavy-tailed noises we injected during tail inflation experiments ($\alpha = 1.4$).

| Task | Beginning | Middle | End |
|---|---|---|---|
| FMNIST, LeNet | 14.3 | 14.2 | 16.5 |
| SVHN, VGG11 | 5.0 | 5.2 | 12.5 |
| CIFAR10, VGG11 | 9.2 | 6.6 | 7.0 |
| dataAug, CIFAR10, VGG11 | 16.2 | 16.2 | 8.5 |
| dataAug, CIFAR100, VGG16 | 35.1 | 14.4 | 5.35 |

