# OpenReview forum: "Eliminating Sharp Minima from SGD with Truncated Heavy-tailed Noise"
_ICLR.cc/2022/Conference — ICLR 2022 Poster_

### Official Review · Reviewer_HGyL · 2021-10-26

**Correctness:** 3
**Technical Novelty And Significance:** 2
**Empirical Novelty And Significance:** 2
**Recommendation:** 5
**Confidence:** 3

**Main Review:**

I appreciate the mathematical rigor of the work, but I am not convinced by its machine learning / deep learning relevance.

Specifically, I find the following points problematic.
1. The title seems inappropriate. The word SGD is taken to mean a special kind of noise that is due to minibatch sampling. This work, however, only studies GD with injected power-law noise. I think it is misleading to say "SGD" in the title

2. The assumptions seem too strong and deep learning irrelevant. One of my main objections is that the paper assumes finitely many minima, yet, neural networks, both underparametrized and overparametrized, should have infinitely many minima, and the fact that the minima of neural networks are degenerate makes it inappropriate to apply a transition graph analysis. This then makes me think that the theoretical analysis is not relevant for deep learning

3. The main result in Theorem 2 only applies to the case when the learning rate is infinitesimal (and the limit needs to be taken under certain scaling conditions). This amounts to a continuous-time approximation, and I think is a crucial limitation of the theoretical results. The theorems reveal nothing about the behavior of SGD at a finite learning rate, which is the actual regime that SGD is run in practice.
- Also, I feel that this continuous-time condition should be stated much earlier in the draft

4. I am unconvinced by the experiment section. Both LeNet and VGG are outdated architectures because they lack the residual structure. I would ask for evaluation on at least a modern ResNet to demonstrate the effectiveness of the proposed method
- Even if evaluated on ResNet, I still think it would not suffice because it is quite easy to improve a vanilla model; I would really want to see being able to improve some state-of-the-art results for the paper to be experimentally convincing

Other important questions (but may not constitute reasons for rejection):
1. It has been found that the power-law index of SGD noise crucially depends on the learning rate of SGD (see https://arxiv.org/abs/2106.02588, or https://arxiv.org/abs/2105.09557). How does this fact affect the theoretical results of this work?


Minor question:
1. page 3: what is "regeneration structure"?

**Summary Of The Paper:**

The paper studies gradient descent with injected power-law tail noise. The work shows that in the infinitesimal learning rate regime, the heavy tail noise can cause GD to not to converge to sharp minima.

Based on this theory, the paper proposes a technique to inject noise to GD to help training


**Summary Of The Review:**

The following two reasons are the main weaknesses, based on which I recommend rejection:
1. The theory seems irrelevant for machine learning / deep learning
2. The experimental evaluation is weak because it uses outdated architecture and because the result only improves on badly performing vanilla training strategies

Therefore, taken as a theoretical paper, I find the theory limited and irrelevant. Taken as an experimental/method paper, I find the improvement and methodology unconvincing.

---

> ### Author Response · Authors · 2021-11-23
> **Authors’ Response to Reviewer HGyL (4/4)**
>
> (Continuing the Above Comment)
>
> **Q: It has been found that the power-law index of SGD noise crucially depends on the learning rate of SGD (see [1,2]). How does this fact affect the theoretical results of this work?**
>
> Regarding the references mentioned by the reviewer, we clarify that what has been established in [1,2] is that the **stationary distribution of the SGD** (or the corresponding continuous-time SDE) can be heavy-tailed. This, however, does not imply that the **gradient noise** is heavy-tailed. In particular, the heavy-tail index (or heavy-tailedness itself) of the gradient noise is not affected by the learning rate.
>
> To be specific, note that the SDEs studied in [1,2] are both driven by **light-tailed** (Gaussian) perturbations. Moreover, [1,2] show that even under **light-tailed** perturbations, heavy tails can arise in the stationary distribution through multiplicative dynamics, and the tail index of the resulting stationary distribution can be characterized by the learning rates. In comparison, we focus on the impact of the heavy-tails **in the gradient noise** and the truncation scheme on the global dynamics of SGD. Our results show that, under heavy-tailed noises, the power-law index $\alpha$ for the tail in noises also characterizes the first exit time and global dynamics of SGD with no dependence on other training hyperparameters. In view of this, it should be clear that our work focuses on a setting that is different from those in [1,2], and our results are compatible with the references mentioned by the reviewer.
>
>
>
> **Q: page 3: what is "regeneration structure"?**
>
> We are referring to the classical concept of renewal/regeneration processes; please see, for example, [26]. For instance, whenever SGD returns to a neighborhood of the local minimum, the SGD process (almost) resets/regenerates itself due to Markov property. In our proof, several observations of this type allow us to partition the SGD trajectory into different renewal/regeneration cycles that are (almost) identically distributed.
>
>
> **References**
>
> [1] Wojtowytsch, Stephan. Stochastic gradient descent with noise of machine learning type. Part II: Continuous time analysis.
>
> [2] Mori, Takashi, et al. Logarithmic landscape and power-law escape rate of SGD.
>
> [3] Dariusz Buraczewsk, Ewa Damek, and Thomas Mikosch. Stochastic Models with Power-Law Tails; The Equation X= AX
>
> [4] Gurbuzbalaban, Mert, Umut Şimşekli, and Lingjiong Zhu. The Heavy-Tail Phenomenon in SGD
>
> [5] Hodgkinson, Liam, and Michael W. Mahoney. Multiplicative Noise and Heavy Tails in Stochastic Optimization
>
> [6] Meng, Qi, et al. Dynamic of stochastic gradient descent with state-dependent noise.
>
> [7]  Şimşekli, Umut, et al. On the Heavy-Tailed Theory of Stochastic Gradient Descent for Deep Neural Networks.
>
> [8] Nguyen, Than Huy, Umut Şimşekli, and Gaël Richard. Non-asymptotic analysis of Fractional Langevin Monte Carlo for non-convex optimization
>
> [9] Şimşekli, Umut, et al. Fractional underdamped langevin dynamics: Retargeting sgd with momentum under heavy-tailed gradient noise.
>
> [10] Camuto, Alexander, et al. Asymmetric Heavy Tails and Implicit Bias in Gaussian Noise Injections.
>
> [11] Garg, Saurabh, et al. On Proximal Policy Optimization's Heavy-tailed Gradients.
>
> [12] Zhang, Jingzhao, et al. Why are adaptive methods good for attention models?
>
> [13] Pavlyukevich, Ilya. Metastable Behaviour of Small Noise Lévy-Driven Diffusion.
>
> [14] Xie, Zeke, Issei Sato, and Masashi Sugiyama. A diffusion theory for deep learning dynamics: Stochastic gradient descent exponentially favors flat minima.
>
> [15] Zhou, Pan, et al. Towards theoretically understanding why sgd generalizes better than adam in deep learning.
>
> [16] Schuss, Zeev, and Bernard J. Matkowsky. The exit problem: a new approach to diffusion across potential barriers.
>
> [17] Holcman, David, and Z. Schuss. Escape through a small opening: receptor trafficking in a synaptic membrane.
>
> [18] Berglund, Nils. Kramers' law: Validity, derivations and generalisations
>
> [19] He, Fengxiang, Tongliang Liu, and Dacheng Tao. Control batch size and learning rate to generalize well: Theoretical and empirical evidence.
>
> [20] Zhu, Zhanxing, et al. The anisotropic noise in stochastic gradient descent: Its behavior of escaping from sharp minima and regularization effects.
>
> [21] Shapiro, A., and Y. Wardi. Convergence analysis of gradient descent stochastic algorithms.
>
> [22] Bubeck, Sébastien. Convex optimization: Algorithms and complexity
>
> [23] Shamir, Ohad, and Tong Zhang. Stochastic gradient descent for non-smooth optimization: Convergence results and optimal averaging schemes
>
> [24] Moulines, Eric, and Francis Bach. Non-asymptotic analysis of stochastic approximation algorithms for machine learning
>
> [25] Hazan, Elad, and Satyen Kale. Beyond the regret minimization barrier: optimal algorithms for stochastic strongly-convex optimization
>
> [26] Ross, Sheldon M., et al. Stochastic processes

---

> > ### Comment · Reviewer_HGyL · 2021-11-24
> > **Reply**
> >
> > Thanks for the update.
> >
> > I feel slightly more positive towards the paper. The paper is indeed more interesting if the most important message is that a heavy-tail noise can eliminate sharp minimum.
> >
> > In this case, I feel that the authors should have spent much more experimental effort in showing that how the sharpness of deep neural networks is controlled/reduced by the proposed heavy-tail noise. The present experiment is not convincing. In particular, it contains too few examples (only 3 experiments at the bottom half of table 1), I feel that much more experiments with much more variations in networks/tasks are required to validate the experiment.
> > * There is also a crucial problem with the experiment that I think is worthy of rejection on its own: there is no error bar/variance reported for the stated numbers in table 1 and 2 -- reporting the variance of experiments should be a minimal standard for acceptance.
> > * One more experiment I think is crucial is to show the dependence of the sharpness (or the expected sharpness) on the hyperparameters that the theory predicts. For example, if the theory predicts an increase of sharpness with an increasing batch size, then the authors should experimentally test this
> >
> > Also (a non-crucial and subjective comment), it is under intensive debate whether sharpness is helpful for generalization, and also I feel that it is both unnecessary and distracting to compare the generalization of the trained networks (since the theory is not about performance).
> >
> > Therefore, while I raise my score to a 5, I still find the paper not convincing for the experiment section and lean towards rejection

---

> > > ### Author Response · Authors · 2021-11-25
> > > **Authors' Response to Reviewer HGyL's Reply (2/2)**
> > >
> > > (Continuing the Above Comment)
> > >
> > > **Q: One more experiment I think is crucial is to show the dependence of the sharpness (or the expected sharpness) on the hyperparameters that the theory predicts. For example, if the theory predicts an increase of sharpness with an increasing batch size, then the authors should experimentally test this**
> > >
> > > We thank the reviewer for the suggestion. In our theoretical results, the most important factors are the truncation threshold $b$ (since the definition of relative width $l^*$ in (eq. 3) and the characterization of sharp/flat minima in our work depend on it) and the heavy-tailed noises (in particular, the power-law index $\alpha$ of the tail). As for $\alpha$, it mostly affects the length of the exit time from a basin (see Theorem 1) and how long it takes for the asymptotics to kick in and phenomenon—the elimination of sharp minima in the trajectory of SGDs— of our interest manifest (see Theorem 2,3). Therefore, an interesting question would be how the choice of truncation threshold $b$ affects the sharpness of the solutions obtained after training. Technically, however, this is difficult to verify experimentally, since our characterization of sharpness relies on $l^*$, the minimum number of jumps needed to visit a different attraction field, and evaluation of this quantity requires the knowledge of the global geometry of loss landscape itself, which is computationally infeasible in deep learning problems. Therefore, for the most part of our experiments, we fix the majority of the training hyperparameters, and focus on how SGD changes with or without clipping and with or without heavy-tailed noises, and it is not immediately clear how metrics such as expected sharpness would depend on the choice of $b$ quantitatively at a very fine granular level.
> > >
> > > That being said, our theoretical analyses do predict the impact of different choices of $b$ qualitatively. In particular, when $b$ is too large, the dynamics would behave very similarly to the heavy-tailed SGD without gradient clipping in the sense that the transitions to basins that are far away will almost always take place by one sudden jump. In view of the poor performance of the heavy-tailed SGDs without gradient clipping (please see our discussion in response to Reviewer kP7Y: https://openreview.net/forum?id=B3Nde6lvab&noteId=goHpAcBnlaH), we can expect that, if $b$ is too large, the heavy-tailed SGD with gradient clipping is likely to find solutions with higher sharpness values (and worse generalization performance). On the other hand, for moderate values of $b$, the theory suggests that the SGDs with gradient clipping are likely to find flatter/wider minima.
> > > In case $b$ is too small, escaping from a relatively sharp/narrow minimum can take very long, and hence the sharpness of the solutions found within a fixed number of iterations may not improve. We are currently running experiments with varying values of $b$ to see if it indeed impacts the sharpness of the solution as predicted above, and we will provide updates once the experiments are done.
> > >
> > >
> > >
> > >
> > >
> > > **Q: Also (a non-crucial and subjective comment), it is under intensive debate whether sharpness is helpful for generalization, and also I feel that it is both unnecessary and distracting to compare the generalization of the trained networks (since the theory is not about performance).**
> > >
> > > We thank the reviewer for this comment. While recent large-scale experiments (see [1] for example) indicate that the sharpness of the solutions does predict their generalization performance under typical deep learning settings, we agree that the current understanding of sharpness (or geometric properties of loss landscape in general) and generalization is limited on both the theoretical and empirical fronts. In the final version of our paper, we will make it clear that the focus of the work is how gradient clipping strongly affects the way heavy-tailed SGD explores and traverses the loss landscape.
> > >
> > >
> > > [1] Yiding Jiang, Behnam Neyshabur, Hossein Mobahi, Dilip Krishnan, and Samy Bengio. Fantastic generalization measures and where to ﬁnd them

---

> > > > ### Comment · Reviewer_HGyL · 2021-11-28
> > > > **final thoughts**
> > > >
> > > > Thanks for the reply. I decide to keep a score of 5. There are two main reasons.
> > > > 1. I feel that the transition graph analysis is not relevant for deep learning. Consider a relu network, every local minimum should extend to infinity due to the rescaling invariance. This means that every local minimum of a neural network is highly non-localized, which in turn means that transition graph analysis is not relevant
> > > > 2. The experiments with neural networks do not convince me: (a) after the update where the authors present the error bars, I find many of the baselines agree with the proposed method in performance given the statistical significance of the result; (b) I think it is also crucial for the authors to study how the sharpnesses of a neural network change as the parameters of training change (and check whether such trends agree with the predictions of the theory).
> > > >
> > > > As high-level comments for future revisions, I feel that the paper may either want to present more convincing experiments to claim relevance to deep learning or remove all discussions about deep learning to avoid overclaim

---

> > > > > ### Author Response · Authors · 2021-11-28
> > > > > **Response (2/2)**
> > > > >
> > > > > **(Continuing the Above Comment)**
> > > > >
> > > > > **Table H1:** Improvements on **test accuracy** in the experiments reported in Table 2 of our main paper.
> > > > >
> > > > > |Task | Our 1 over SB | Our 1 over SB + Clip | Our 2 over SB |Our 2 over SB + Clip |
> > > > > |--|--|--|--|--|
> > > > > |FMNIST,LeNet | 0.27%$\pm$0.73% | 0.7%$\pm$0.64% | 0.86%$\pm$0.97%| 1.3%$\pm$0.81% |
> > > > > |SVHN,VGG11 | 2.5%$\pm$0.34%| 2.48%$\pm$0.30% | 2.44%$\pm$0.39% | 2.42%$\pm$0.38%|
> > > > > |CIFAR10,VGG11 | 1.27%$\pm$0.91% |1.31%$\pm$1.50% |1.45%$\pm$0.51%|1.49%$\pm$0.86%|
> > > > >
> > > > > **Table H2**: Improvements on **sharpness** in the experiments reported in Table 2 of our main paper
> > > > >
> > > > > |Task | Our 1 over SB | Our 1 over SB + Clip | Our 2 over SB |Our 2 over SB + Clip |
> > > > > |--|--|--|--|--|
> > > > > | FMNIST,LeNet | -0.005$\pm$0.0009 | -0.006$\pm$0.0009 | -0.006$\pm$0.001 | -0.006$\pm$0.001 |
> > > > > | SVHN,VGG11 | -0.035$\pm$0.007 | -0.039$\pm$0.0095 | -0.032$\pm$0.009 | -0.032$\pm$0.009 |
> > > > > |CIFAR10,VGG11 | -0.026$\pm$0.009 | -0.014$\pm$0.022 | -0.012$\pm$0.010 | -0.013$\pm$0.022|
> > > > >
> > > > > **Table H3**: Improvements on sharpness and test accuracy in the experiments reported in Table 2 of our main paper. We only use SB + Clip as baseline because without gradient clipping, the model weights often explode with the vanilla SB method.
> > > > >
> > > > > |  | Sharpness, Our 1 |  Sharpness, Our 2 | Accuracy, Our 1 | Accuracy, Our 2 |
> > > > > |--|--|--|--|--|
> > > > > | CIFAR10, VGG11 | -0.082$\pm$0.004 | -0.071$\pm$0.006 | 1.15%$\pm$0.28% | 0.91%$\pm$0.28%|
> > > > > | CIFAR100, VGG16 | -0.416$\pm$0.053 | -0.379$\pm$0.085 | 9.12%$\pm$1.45% |6.67%$\pm$2.76%|
> > > > >
> > > > > **Q3: (b) I think it is also crucial for the authors to study how the sharpnesses of a neural network change as the parameters of training change (and check whether such trends agree with the predictions of the theory).**
> > > > >
> > > > > As suggested by the reviewer, we conducted new experiments where we vary the gradient clipping threshold $b$ in our method and see if it aligns (qualitatively) with the prediction based on our theoretical results. In particular, we focus on the VGG11+CIFAR10 setting in the ablation study. Results are shown in the table below.
> > > > >
> > > > > In this response we focus on the change on expected sharpness, as suggested by the reviewer.
> > > > > First, under extremely small $b$, escaping from even a small basin may require a large number of jumps, resulting in significantly extended exit time as predicted by Theorem 1. Therefore, as shown in the table, truncated heavy-tailed SGD may be trapped by a sharp minima.
> > > > > Next, when $b$ is too large, truncated heavy-tailed SGD almost degenerates to the unclipped version, thus inheriting the instability and the deteriorated performance of unclipped heavy-tailed SGD shown in our ablation study. In particular, the sharpness value and training/test accuracy of the obtained solution is worse than that of (unclipped) vanilla SGD. As shown in the table below, this is indeed the case. Moreover, we mention that with very large $b$ (such as $b = 10$ and $b = 25$), the training accuracy in all of the 5 replications becomes 10% (completely random level). Since these cases become rather uninformative, we didn’t include them in the table, but we reiterate that when the clipping threshold is too large, the truncated heavy-tailed method fails to locate and converge to a local minimum that performs well even on the training set.
> > > > >
> > > > > **Table H4:** Improvements (mean and 95% CI over 5 reps) on sharpness and test accuracy for CIFAR10-VGG11 over different clipping thresholds:
> > > > >
> > > > > |b|0.005|0.025|0.1|0.25|0.5|1| 2.5| 5|
> > > > > |--|--|--|--|--|--|--|--|--|
> > > > > | Expected Sharpness | 1.4$\pm$0.026 | 1.39$\pm$0.107 | 0.205$\pm$0.016 | 0.008$\pm$0.0005 | 0.006$\pm$0.0005 | 0.114$\pm$0.004 | 0.518$\pm$0.454 | 0.596$\pm$0.523 |
> > > > > | Test Accuracy | 70.20%$\pm$0.26% | 71.05%$\pm$0.43% | 74.57%$\pm$0.39% | 76.87%$\pm$0.24% | 76.91$\pm$0.35% | 76.33%$\pm$0.24% | 52.53%$\pm$20.96% | 30.39%$\pm$24.53%|
> > > > >
> > > > >
> > > > >
> > > > > [1] Neyshabur, Behnam, et al. Exploring generalization in deep learning
> > > > >
> > > > > [2] Zhu, Zhanxing, et al. The anisotropic noise in stochastic gradient descent: Its behavior of escaping from sharp minima and regularization effects.

---

> > > > > > ### Author Response · Authors · 2021-11-28
> > > > > > **Response: Raw Data (2/2)**
> > > > > >
> > > > > > **DataAug, CIFAR10, Accuracy**
> > > > > >
> > > > > > |rep|1|2|3|4|5|
> > > > > > |--|--|--|--|--|--|
> > > > > > |SB + Clip|89.40%|89.41%|89.89%|89.52%|89.47%|
> > > > > > |Our 1|90.76%|90.57%|90.49%|90.85%|90.79%|
> > > > > > |Our 2|90.67%|90.23%|90.52%|90.13%|90.70%|
> > > > > > |increment our 1|1.36%|1.16%|0.60%|1.33%|1.32%|
> > > > > > |increment our 2|1.27%|0.82%|0.63%|0.61%|1.23%|
> > > > > >
> > > > > > **DataAug, CIFAR10, Sharpness**
> > > > > >
> > > > > > |rep|1|2|3|4|5|
> > > > > > |--|--|--|--|--|--|
> > > > > > |SB+Clip|0.16452637|0.17151361|0.16145924|0.17511394|0.16216042
> > > > > > |Our 1|0.08815541|0.08848491|0.08129915|0.08799554|0.07889951|
> > > > > > |Our 2|0.09454982|0.0915722|0.0991595|0.10010046|0.0966791|
> > > > > > |increment our 1|-0.07637096|-0.0830287|-0.08016009|-0.0871184|-0.0832609|
> > > > > > |increment our 2|-0.06997655|-0.07994141|-0.06229974|-0.07501348|-0.06548132|
> > > > > >
> > > > > > **DataAug, CIFAR100, Accuracy**
> > > > > >
> > > > > > |rep|1|2|3|4|5|
> > > > > > |--|--|--|--|--|--|
> > > > > > |SB + Clip|55.76%|56.80%|56.38%|56.35%|56.32%|
> > > > > > |Our 1|67.43%|65.12%|65.14%|65.96%|63.57%|
> > > > > > |Our 2|67.19%|61.17%|60.97%|64.75%|60.90%|
> > > > > > |increment our 1|11.67%|8.32%|8.76%|9.61%|7.25%|
> > > > > > |increment our 2|11.43%|4.37%|4.59%|8.40%|4.58%|
> > > > > >
> > > > > > **DataAug, CIFAR100, Sharpness**
> > > > > >
> > > > > > |rep|1|2|3|4|5|
> > > > > > |--|--|--|--|--|--|
> > > > > > |SB + Clip|0.86552121|0.84463337|0.83883204|0.88976819|0.8479574|
> > > > > > |Our 1|0.35236142|0.44577675|0.45834895|0.46022679|0.48895717|
> > > > > > |Our 2|0.34967315|0.50547866|0.53556461|0.44756501|0.55534815|
> > > > > > |increment our 1|-0.51315979|-0.39885662|-0.38048309|-0.4295414|-0.35900023|
> > > > > > |increment our 2|-0.51584806|-0.33915471|-0.30326743|-0.44220318|-0.29260925|

---

> > > > > > ### Author Response · Authors · 2021-11-28
> > > > > > **Response: Raw Data (1/2)**
> > > > > >
> > > > > > **Table: Corrupted FMNIST, accuracy**
> > > > > >
> > > > > > | rep | SGD | SGD + Clip | Our 1 | Our 2 | Our 1 increment over SB | Our 1 increment over SB + Clip | Our 2 increment over SB | Our 2 increment over SB + Clip |
> > > > > > |--|--|--|--|--|--|--|--|--|
> > > > > > | 1 | 68.72% | 68.46% | 70.16% | 69.91% | 1.44% | 1.70% | 1.19% | 1.45% |
> > > > > > | 2 | 70.31% | 69.49% | 70.46% | 70.40% | 0.15% | 0.97% | 0.09% | 0.91% |
> > > > > > | 3 | 68.82% | 68.35% | 68.23% | 69.84% | -0.59% | -0.12% | 1.02% | 1.49% |
> > > > > > | 4 | 68.17% | 68.04%| 68.91% | 70.62% | 0.74% | 0.87% | 2.45% | 2.58% |
> > > > > > | 5 | 69.97% | 69.49%| 69.59% | 69.54% | -0.38% | 0.10% | -0.43% | 0.05% |
> > > > > >
> > > > > >
> > > > > >
> > > > > > **Table: Corrupted FMNIST, Sharpness**
> > > > > >
> > > > > > | rep | SGD | SGD + Clip | Our 1 | Our 2 | Our 1 increment over SB | Our 1 increment over SB + Clip | Our 2 increment over SB | Our 2 increment over SB + Clip |
> > > > > > |--|--|--|--|--|--|--|--|--|
> > > > > > | 1 | 0.0063839 | 0.00731466 | 0.00259681 | 1.43E-03 | -0.00378709 | -0.00471785 | -4.95E-03 | -5.88E-03 |
> > > > > > | 2 | 0.00697866 | 0.00794059 | 0.00265279 | 1.77E-03 | -0.00432587 | -0.0052878 | -5.21E-03 | -6.17E-03 |
> > > > > > | 3 | 0.00954565 | 0.00983982 | 0.0034883 | 2.13E-03 |  -0.00605735 | -0.00635152 | -7.42E-03 | -7.71E-03 |
> > > > > > | 4 | 0.00918573 | 0.00990952 | 0.00318139 | 1.78E-03 | -0.00600434 | -0.00672813 | -7.41E-03 | -8.13E-03 |
> > > > > > | 5 | 0.00694936 | 0.0101638 | 0.0028609 | 2.00E-03 | -0.00408846 | -0.0073029 | -4.95E-03 | -8.16E-03 |
> > > > > >
> > > > > >
> > > > > > **Table: SVHN, Test Accuracy**
> > > > > >
> > > > > > | rep | SGD | SGD + Clip | Our 1 | Our 2 | Our 1 increment over SB | Our 1 increment over SB + Clip | Our 2 increment over SB | Our 2 increment over SB + Clip |
> > > > > > |--|--|--|--|--|--|--|--|--|
> > > > > > | 1 | 86.22% | 86.21% |88.57% | 87.98% | 2.35% | 2.36% | 1.76% | 1.77%|
> > > > > > | 2 | 85.63% | 85.72% | 88.56% |88.26% | 2.93% | 2.84% | 2.63% | 2.54%
> > > > > > | 3 | 85.69% | 85.69% | 88.54% | 88.66% | 2.85% | 2.85% | 2.97% | 2.97% |
> > > > > > | 4 | 85.92% | 86.08% | 88.28% | 88.37% | 2.36% | 2.20% | 2.45% | 2.29% |
> > > > > > | 5 | 86.16% | 86.04% | 88.17% | 88.57% | 2.01% | 2.13% | 2.41% | 2.53% |
> > > > > >
> > > > > > **Table: SVHN, Sharpness**
> > > > > >
> > > > > > | rep | SGD | SGD + Clip | Our 1 | Our 2 | Our 1 increment over SB | Our 1 increment over SB + Clip | Our 2 increment over SB | Our 2 increment over SB + Clip |
> > > > > > |--|--|--|--|--|--|--|--|--|
> > > > > > | 1 | 0.02939579 | 0.04335717 | 0.00277029 | 0.01202298 | -0.0266255 | -0.04058688 | -1.74E-02 | -3.13E-02 |
> > > > > > | 2 | 0.04278039 | 0.03904421 | 0.00153741 | 0.00415514 | -0.04124298 | -0.0375068 | -3.86E-02 | -3.49E-02 |
> > > > > > | 3 | 0.03312451 | 0.03913923 | 0.00153741 | 0.00290837 | -0.0315871 | -0.03760182 | -3.02E-02 | -3.62E-02 |
> > > > > > | 4 | 0.03297316 | 0.03292073 | 0.00351841 | 0.0034521 | -0.02945475 | -0.02940232 | -2.95E-02 | -2.95E-02 |
> > > > > > | 5 | 0.04704038 | 0.05003312 | 0.00207545 | 0.00190536 | -0.04496493 | -0.04795767 | -4.51E-02 | -4.81E-02 |
> > > > > >
> > > > > > **Table: Ablation study, CIFAR10 VGG11, Test Accuracy**
> > > > > >
> > > > > > | rep | SGD | SGD + Clip | Our 1 | Our 2 | Our 1 increment over SB | Our 1 increment over SB + Clip | Our 2 increment over SB | Our 2 increment over SB + Clip |
> > > > > > |--|--|--|--|--|--|--|--|--|
> > > > > > | 1 | 74.19% | 74.52% | 73.63% | 76.29% | -0.56% | -0.89% | 2.10% | 1.77% |
> > > > > > | 2 | 75.05% | 74.60% | 76.71% | 76.93% | 1.66% | 2.11% | 1.88% | 2.33% |
> > > > > > | 3 |  74.07% | 75.04% | 75.93% | 75.53% | 1.86% | 0.89% | 1.46% | 0.49% |
> > > > > > | 4 | 74.07% | 74.83% | 75.57% | 75.24% | 1.50% | 0.74% | 1.17% | 0.41% |
> > > > > > | 5 | 74.72% | 72.89% | 76.61% | 75.34% | 1.89% | 3.72% | 0.62% | 2.45%|
> > > > > >
> > > > > > **Table: Ablation study, CIFAR10 VGG11, Sharpness**
> > > > > >
> > > > > > | rep | SGD | SGD + Clip | Our 1 | Our 2 | Our 1 increment over SB | Our 1 increment over SB + Clip | Our 2 increment over SB | Our 2 increment over SB + Clip |
> > > > > > |--|--|--|--|--|--|--|--|--|
> > > > > > |1|0.048396|0.028607|0.030668|0.026583|-0.017728|0.002061|-2.18E-02|-2.02E-03|
> > > > > > |2|0.043174|0.024456|0.018279|0.035966|-0.024895|-0.006177|-7.21E-03|1.15E-02|
> > > > > > |3|0.068886|0.027447|0.027202|0.04182|-0.041684|-0.000245|-2.71E-02|1.44E-02|
> > > > > > |4|0.056959|0.036203|0.027679|0.049727|-0.02928|-0.008524|-7.23E-03|1.35E-02|
> > > > > > |5|0.030907|0.076229|0.016846|0.032558|-0.014061|-0.059383|1.65E-03|-4.37E-02|

---

> > > > > ### Author Response · Authors · 2021-11-28
> > > > > **Response (1/2)**
> > > > >
> > > > > We thank the reviewer for the reply and we want to make the following three clarifications.
> > > > >
> > > > > **Q1: I feel that the transition graph analysis is not relevant for deep learning. Consider a relu network, every local minimum should extend to infinity due to the rescaling invariance. This means that every local minimum of a neural network is highly non-localized, which in turn means that transition graph analysis is not relevant**
> > > > >
> > > > > Thank you for this question. We first point out that in practically all real-world supervised learning tasks, RELU (or any other activation function for that matter) is paired with $L_2$ regularization, and in such a case, **local minima do not extend to infinity**.
> > > > >
> > > > > More importantly, even without $L_2$-norm penalty, connected local minima (possibly extending to infinity) **do not invalidate the concepts of attraction fields or the transition graph structure between these attraction fields**: as we have explained in our previous comment, the concepts we introduced can be easily generalized and the attraction fields are, in general, defined for different manifolds of connected local minima (possibly extending to infinity), see https://openreview.net/forum?id=B3Nde6lvab&noteId=z-y7Xlijco.
> > > > >
> > > > > **Q2: ​​The experiments with neural networks do not convince me: (a) after the update with the error bars, I find many of the baselines agree with the proposed method in performance given the statistical significance of the result**
> > > > >
> > > > > Thank you for checking our updated tables. However, we believe that the improvement of **our method is consistent and “statistically” significant in most cases**, and for the few exceptions, it is either due to the special nature of the task at hand or can be easily resolved by tuning the hyperparameters; please see the bullet points below. Moreover, we believe that simply checking if the “confidence intervals” overlap in the presented table is not a reliable way to decide whether the improvements are significant or not. In particular, because the experiments for different methods are not independent—instead, within each replication of our experiments, the same initial model weights are shared by all the different methods we tested (for example, LB, SB, SB+Cip, SB+Noise, Our1, Our2 in our ablation study)—and hence, a much more reliable way to test the significance of the improvement is to *compute the differences (i.e., improvement)* in the outcome (i.e, test accuracy and sharpness) to estimate the improvements and consider the *confidence intervals of the estimated improvements*.
> > > > >
> > > > > Below, we present the improvements induced by our methods, which is a more informative and definitive metric (compared to considering different methods independently and simply checking if the confidence intervals overlap). In the tables below, we can see that the improvements are statistically significant in most cases, and there are only a few cases where the confidence interval overlaps with 0. We make the following remarks:
> > > > >
> > > > > * In the FMNIST task (‘FMNIST,LeNet’ in the first rows of Table H1 and H2 below), the training data is corrupted by randomly switching the labels of a certain portion of the training data. It has been well documented that, for loss landscapes induced by partially corrupted data, the flatness of local minima does not necessarily indicate strong test accuracy (see [1] for example). Therefore, we should only focus on the sharpness results in this case, and the reported test accuracy should be disregarded: the fact that the confidence interval includes 0 should not be considered as evidence to dismiss the improvement due to the injection of truncated heavy-tails.
> > > > > * On the other hand, the improvement in sharpness is clear in this task as our theory predicts.
> > > > > Other than the test accuracy of the corrupted FMNIST task, there is only one case where the improvement is not significant: the VGG11,CIFAR10 task in the ablation study. Considering the stochastic nature of the algorithms and the fact that the tasks and training hyperparameters are largely adapted from those in [2] for fair comparison (rather than fine-tuned for our method), the magnitude and consistency in the improvement of test accuracy and expected sharpness under our method is clearly significant. In fact, in the experiment with varying truncation threshold $b$ (see our new results in Table H4), we see that simply picking different thresholds raises the test accuracy to 76.91$\pm$0.35% (as opposed to 74.42%$\pm$0.40% and 74.38%$\pm$0.77% in the baselines SB and SB+Clip, respectively) and lowers the sharpness to 0.006$\pm$0.0005 (as opposed to 0.050$\pm$0.013 and 0.039$\pm$0.019  in the baselines) leading to clearly significant improvements in both measures. (In comparison, we observed that tuning $b$ in SGD + Clip won't lead to improvements in either accuracy or sharpness.)
> > > > >
> > > > > For detailed numbers, please see the raw data attached at the end.
> > > > >
> > > > > **(To Be Continued in the Comment Below)**

---

> > > ### Author Response · Authors · 2021-11-25
> > > **Authors' Response to Reviewer HGyL's Reply (1/2)**
> > >
> > > We thank the reviewer for the prompt feedback.
> > >
> > > **Q: I feel that the authors should have spent much more experimental effort in showing that how the sharpness of deep neural networks is controlled/reduced by the proposed heavy-tail noise. The present experiment is not convincing. In particular, it contains too few examples (only 3 experiments at the bottom half of table 1), I feel that much more experiments with much more variations in networks/tasks are required to validate the experiment.**
> > >
> > > While we agree that our work would benefit from a wider range of experiments, we want to point out that there are experiments in our paper that the reviewer seems to have overlooked. More specifically, we evaluated the sharpness for the experiments in Table 2 (which presents our experimental results on data augmentation + learning rate scheduling training tasks) as well and observed the same pattern: flatter solution with better generalization performance under our heavy-tailed method. Please see Table A.3 in the appendix. This means that we have verified the effect of truncated heavy-tailed SGD and the validity of our theoretical results in at least 5 different deep learning examples (rather than only 3).
> > >
> > > On a related note, we also present two other sharpness measures (Maximal sharpness and PAC-Bayes sharpness) in Appendix A.3 for all five examples, and the results are very consistent with the ones w.r.t. the expected sharpness.
> > >
> > > In the final version of our paper, we will present all the sharpness results for all five examples in the main paper.
> > >
> > > **Q: There is also a crucial problem with the experiment that I think is worthy of rejection on its own: there is no error bar/variance reported for the stated numbers in table 1 and 2 -- reporting the variance of experiments should be a minimal standard for acceptance.**
> > >
> > > We did not include error bar/variance in Table 1 and 2 only because (i) the replication count 5 is too small to warrant the validity of the normal approximation to the data for constructing confidence intervals, which makes error bars potentially misleading, and hence, instead, (ii) it would be more informative and straightforward to provide the raw data in all 5 replications, especially considering the consistency of the improvements (please see, for instance, Fig A.6 in Appendix).
> > >
> > > The details such as the sharpness values for experiments in Table 2 and the raw data of 5 replications are presented in the appendix due to space limit. In the two tables below, we provide the means and the 95% CI’s for expected sharpness and test performance for all deep learning experiments in our paper. We will update all the tables and provide CI in the final version of our paper.
> > >
> > > *(Table 1 updated) Test accuracy and expected sharpness in the ablation study. Mean and 95% CI over 5 replications.*
> > >
> > > | **Test Accuracy** | LB | SB | SB + Clip | SB + Noise | Our 1 | Our 2|
> > > |--|--|--|--|--|--|--|
> > > | FMNIST, LeNet| 68.66%$\pm$0.41% | 69.20%$\pm$0.79% | 68.77%$\pm$0.59% | 64.43%$\pm$3.37% | 69.47%$\pm$0.8% | **70.06%$\pm$0.38%** |
> > > | SVHN, VGG11 | 82.87%$\pm$0.39% | 85.92%$\pm$0.23% | 85.95%$\pm$0.2% | 38.85%$\pm$24.13% | **88.42%$\pm$0.16%** | 88.37%$\pm$0.24% |
> > > | CIFAR10, VGG11 | 69.39%$\pm$0.49% | 74.42%$\pm$0.40% | 74.38%$\pm$0.77% | 40.50%$\pm$25.07% | 75.69%$\pm$1.11% | **75.87%$\pm$0.65%** |
> > >
> > > | Expected Sharpness | LB |SB | SB + Clip | SB + Noise | Our 1 | Our 2 |
> > > |--|--|--|--|--|--|--|
> > > | FMNIST, LeNet | 0.032$\pm$0.006 | 0.008$\pm$0.001 | 0.009$\pm$0.001 | 0.047$\pm$0.02 | 0.003$\pm$0.0003 | **0.002$\pm$0.0002** |
> > > | SVHN, VGG11 | 0.694$\pm$0.048 | 0.037$\pm$0.007 | 0.041$\pm$0.006 | 0.012$\pm$0.009 | **0.002$\pm$0.0007** | 0.005$\pm$0.004 |
> > > | CIFAR10, VGG11 | 2.043$\pm$0.083 | 0.050$\pm$0.013 | 0.039$\pm$0.019 | 2.046$\pm$2.4 | **0.024$\pm$0.005** | 0.037$\pm$0.007|
> > >
> > > *(Table 2 updated) Test accuracy and expected sharpness in the data augmentation + LR scheduling tasks. Mean and 95% CI over 5 replications.*
> > >
> > > | CIFAR10, VGG11 | Test Accuracy | Expected Sharpness |
> > > |--|--|--|
> > > | SB + Clip | 89.54%$\pm$0.18% | 0.167$\pm$0.005 |
> > > | Our 1 | 90.67%$\pm$0.13% |0.085$\pm$0.004 |
> > > | Our 2 | 90.45%$\pm$0.23% | 0.096$\pm$0.003 |
> > >
> > > | CIFAR100, VGG16 | Test Accuracy | Expected Sharpness |
> > > |--|--|--|
> > > | SB + Clip | 56.32%$\pm$0.32% | 0.857$\pm$0.018 |
> > > | Our 1 | 65.44%$\pm$1.23% | 0.441$\pm$0.046 |
> > > | Our 2 | 62.99%$\pm$2.5% | 0.479$\pm$0.073 |
> > >
> > > **(To Be Continued in the Comment Below)**

---

> > > ### Author Response · Authors · 2021-11-29
> > > **On Reviewer HGyL's final comment posted on Nov 28 below (Title: final thoughts)**
> > >
> > > Regarding the reviewer's latest comment posted on Nov 28 below (Title: final thoughts), we have thoroughly addressed the questions and concerns the reviewer raised. Since our comments show only upon clicking on 'View 4 More Replies →' button due to the depth of the thread, we provide hyperlinks to our comments here along with a brief summary:
> > >
> > > * The reviewer questions our paper's relevance to deep learning because the reviewer is concerned if our analysis is incompatible with the scale invariance of RELU activation functions whose local minima extend to infinity. However, we clarify that under $L_p$ regularization, which is almost always used in real-world deep learning tasks, local minima won't extend to infinity. More importantly, even without $L_p$ regularization, the concepts in our work do generalize to loss landscapes with connected local minima (possibly extending to infinity). Therefore, our work is still relevant to deep learning.
> > > * In terms of both test accuracy and sharpness, the improvement of our method is highly consistent and statistically significant in most cases. The few exceptions are either due to the special nature of the task at hand (and hence, are as expected) or can be resolved by a very mild parameter tuning in our algorithm.
> > > * We have conducted extra experiments suggested by the reviewer and discussed how the results are aligned with our theories.
> > >
> > > For details, please see our responses.
> > >
> > > (1/2) https://openreview.net/forum?id=B3Nde6lvab&noteId=tsCI4SnP8W1
> > >
> > > (2/2) https://openreview.net/forum?id=B3Nde6lvab&noteId=Hkg1Jsa3Gdr

---

> ### Author Response · Authors · 2021-11-23
> **Authors’ Response to Reviewer HGyL (3/4)**
>
> **(Continuing the Above Comment)**
>
> We also point out that asymptotic analysis (similar to ours in terms of the form) is very common in existing works. For example,  [1,2], the two references mentioned by the reviewer, carry out the first exit time analyses under the classical Kramer’s time approach. However, when applied to the discrete-time SGD (rather than the continuous time SDEs in [1,2]), Kramer-time approach only yields an *asymptotic* result about how the exit time exponentially scales with $\eta$ as it becomes *infinitesimally small*; see, for instance, [18]. Moreover, one major contribution in [1,2] is the explicit expression of the stationary distribution of the SDE, which technically is only attained after running the dynamics for *infinitely long time*. Similarly, latest works such as [1,2,4,5,6,7,9,10,13,14] all resort to asymptotic analysis, but we believe that the techniques and insights established in and indicated by this highly active line of researches provide meaningful insights that facilitate our understanding of SGD in machine learning, and we cannot disregard them as irrelevant simply because of their asymptotic nature.
>
> Lastly, we point out that the scalings in our theoretical results also have important and concrete implications in the behavior of SGD. For example, the scale of the time horizon $O(1/\eta^\beta)$ in Theorem 2 is a polynomial with a moderate degree. This means that the asymptotics will kick in within polynomial time, and hence, we can expect to observe the limit behavior (i.e., elimination of sharp minima) within a realistic training time horizon. This is in sharp contrast to the exponential scaling of the first exit time that arises in the classical metastability analysis of SGDs with light-tailed Langevin dynamics. For example, we can observe in Figure 1 that the asymptotics in Theorem 2 is clearly in force within 3M iterations, whereas the first exit time of the light-tailed SGD never materializes (i.e., SGD does not manage to escape from local minimum even once) since the first exit time increases at an exponential rate and is expected to be of order $10^{450}$. In other words, the particular scaling that appears in our analysis is not arbitrary or just for the convenience of the analysis, but it characterizes important aspects of the global dynamics we analyze.
>
>
> **Q: I am unconvinced by the experiment section. Both LeNet and VGG are outdated architectures because they lack the residual structure. I would ask for evaluation on at least a modern ResNet to demonstrate the effectiveness of the proposed method
> Even if evaluated on ResNet, I still think it would not suffice because it is quite easy to improve a vanilla model; I would really want to see being able to improve some state-of-the-art results for the paper to be experimentally convincing**
>
> We appreciate the reviewer’s suggestion on numerical experiments. We ran extra experiments on WideResNet and observed results that are consistent with the other experiments: the SGD’s with inflated and truncated heavy-tails consistently outperforms the vanilla SGD on WRN.
>
> *Table: WRN28_10, Data Augmentation + Cosine LR scheduling, Test error (percentage) averaged over 5 replications*
>
> |  Task | SGD  | Our  |
> |---|---|---|
> |  CIFAR10, Cutout |  3.5 | **3.35**  |
> |  CIFAR100, Cutout | 20.31  | **19.62**   |
> | CIFAR10, AutoAug  | 2.56  | **2.48**  |
> | CIFAR100, AutoAug  | 17.79  | **17.49**  |
>
> While our design and scale of experiment is as thorough as any other latest theoretical works about noises in SGD and effects on generalization or exit times—such as [1,2,4,5,6,8,9,10,14,15,19,20]—we admit that our experiments are not particularly large-scale to pass for an empirical paper. In particular, while we leave open the possibility that the tail-inflation scheme can contribute to achieving the state-of-the-art performance, we leave such implementation as a future work because the focus of our current work is
> * to establish a principled approach to theoretical analysis of the global dynamics of SGD; and
> * shed light on the impact of the heavy-tails and gradient clipping in the sharpness and the generalization performance of the solutions found by SGD,
>
> rather than designing or implementing a state-of-the-art training algorithm. We clarify that the purpose of our deep learning experiments in the current work is to confirm that the phenomena (elimination of sharp minima) we analyze here are indeed present in high-dimensional deep learning settings, and the theory developed in our paper is relevant to deep learning, through controlled experiments (i.e., ablation study).
>
> The publication of this paper and the dissemination of the results herein will make the resource and attention required for the optimal implementation and extensive parameter tuning for the tail-inflation scheme possible and facilitate much more thorough and larger-scale experiments.
>
> **(To Be Continued in the Comment Below)**

---

> ### Author Response · Authors · 2021-11-23
> **Authors’ Response to Reviewer HGyL (2/4)**
>
> **(Continuing the Above Comment)**
>
> **(ii) Multiple-jump principle and connected local minima**
>
> While we agree that the existence of connected local minima will complicate our analyses, we believe that our technique for characterizing the basin-hopping dynamics and the phenomena we investigate—elimination of sharp minima—associated with the truncated heavy-tailed SGD still carry over to the general setting with connected local minima. Specifically, we modify the definition of attraction fields as follows. First, we consider the set of all the local minima and partition it into a finite number of sets of connected minima. Then for each of such connected regions of minima, its attraction field is defined as all the points from which the ODE flow (with no jump) converges to this connected region. Note that the isolated local minima will become singletons under this partition.
>
> Under small learning rates, the dynamics of the scaled SGD (with truncated heavy-tailed noise) is expected to resemble that of the reflected Brownian motion (or Levy processes) on the manifold of the connected set of minima until some large perturbations send the SGD out. Once the SGD iterates leave the connected minima and attempt to escape from the current attraction field, the exit time and path will again be characterized by $l^*$, the minimum number of jumps to escape, resulting in much longer time spent at basins with higher $l^*$’s and near elimination of the proportion of time spent in those with small $l^*$’s. We also emphasize that this is confirmed in our experiments: note that the loss landscape in our experiment in Fig. 3 (see Section 3) has connected local minima in basin $\Omega_2$, and elimination of sharp minima is still observed. In summary, we believe that there is strong evidence suggesting that the gist of our analysis—i.e. the first exit time analysis and the elimination of sharp minima under gradient clipping—is still valid on general landscapes with connected minima.
>
> **Q: Theorem 2 only applies to the case when learning rate is infinitesimal (and under certain scaling conditions). This amounts to a continuous-time approximation, and I think is a crucial limitation of the theoretical results. The theorems reveal nothing about the behavior of SGD at a finite learning rate, which is the actual regime that SGD is run in practice. Also, this continuous-time condition should be stated much earlier in the draft**
>
> Thank you for raising this question. Let us start by reiterating that our analysis, in fact, does not involve any approximation of SGD with continuous-time processes. Instead, we directly analyze the asymptotics of SGD (unlike many other works in the literature that resort to the approximation of the SGD with small noise stochastic differential equations). In fact, this is one of the significant contributions of our new analysis.
>
> On the other hand, we do acknowledge the asymptotic nature of our analysis: for example, Theorem 2 is indeed an *asymptotic* result, meaning that it describes the behavior of SGD when the learning rate converges to zero. We will make this clear in the final version of our paper. However, *this does not mean that our analysis is irrelevant to the behavior of SGD with finite learning rates*. Instead, Theorem 2 dictates what the system behavior will resemble—i.e., the RHS of (5)—when the learning rate is small but finite.
>
> Asymptotic analysis is a tested and proven approach in many engineering and science contexts, serving as powerful means to study the behaviors of complex systems that are intractable otherwise. However, we admit that its interpretation and application require careful examination. For example, in Theorem 2, although the convergence (5) guarantees that, with a small enough learning rate, the fraction of the time SGD spends at sharp local minima will become near zero; a critical question is how small is small enough. Does the learning rate really have to be infinitesimally small to a degree that the asymptotics (5) becomes irrelevant to the practical learning rates used in deep learning tasks? Or does the asymptotics kick in reasonably early, and SGD with a reasonably small learning rate already behaves similarly to the limit on the RHS of (5), providing meaningful intuition to the global dynamics of SGD in deep learning? To address this question, we conducted our experiments on synthetic loss landscapes (Section 3) and benchmark deep learning tasks (Section 4). In these experiments, one can observe that the global behavior predicted in Theorem 2—i.e., the elimination of sharp minima and the tendency to find flatter minima with better generalization—are present for reasonable / typical learning rates $\eta$ (or a typical scheduling/annealing of $\eta$) in the synthetic examples and deep learning tasks. We emphasize that the learning rates used in our experiments are *not infinitesimal*; please see table A.1 and Section A.4.
>
> **(To Be Continued in the Comment Below)**

---

> ### Author Response · Authors · 2021-11-23
> **Authors’ Response to Reviewer HGyL (1/4)**
>
> We thank the reviewer HGyL for careful reading and asking detailed questions. Before we address the specific concerns of the reviewer, we emphasize that the focus of this paper is to establish a theoretical foundation for understanding the global dynamics of heavy-tailed SGD and its implication in SGD’s ability to avoid sharp local minima and generalization performance.
>
> We acknowledge that we make simplifying assumptions (such as finitely many local minima and asymptotic limits), but our assumptions are modest compared to the other recent theoretical works on the same subject, and we show that the phenomena predicted by the theory persists under much more general settings through carefully designed experiments.
>
> We clarify that although we propose new training schemes based on our theory and test them in deep learning contexts, our purpose is not to achieve state-of-the-art performance. Rather, our goal is to confirm that the phenomena (elimination of sharp minima) our theory predicts are indeed present in high-dimensional deep learning settings through controlled experiments (i.e., ablation study).
>
>
>
>
>
> **Q: The title seems inappropriate. The word SGD is taken to mean a special kind of noise that is due to minibatch sampling. This work, however, only studies GD with injected power-law noise. I think it is misleading to say "SGD" in the title.**
>
> Thank you for the suggestion. However, we respectfully point out that the term “stochastic gradient descent” has been used in a much broader sense than just minibatch sampling in the stochastic optimization and general machine learning contexts. For example, as early as 1996, “stochastic gradient descent” had been used to refer to a broad class of first-order methods under very general stochastic gradient oracles; please see [21], for example. Textbooks such as [22] also differentiate between basic SGD and “mini-batch SGD.” Moreover, in recent machine learning literature, there is a plethora of well-established works where the authors use SGD to refer to a general class of first-order method with arbitrary stochastic noise distributions and does not mention mini batches (see, for instance, [23-25]). While we agree that mini-batch sampling from a given training data pool is a standard approach to produce stochastic gradients in machine learning tasks, the existing works on SGD have been discussing much more abstract mechanisms of accessing stochastic gradients and analyzing a much wider class of first order algorithms. In our work, we focus on the important yet less-studied case where the gradient noise distributions have heavy tails. This setting is well aligned with the conventions and definitions of SGD in the aforementioned works.
>
> **Q: The assumptions seem too strong and deep learning irrelevant. One of my main objections is that the paper assumes finitely many minima, yet, neural networks, both underparametrized and overparametrized, should have infinitely many minima, and the fact that the minima of neural networks are degenerate makes it inappropriate to apply a transition graph analysis. This then makes me think that the theoretical analysis is not relevant for deep learning**
>
> We appreciate the reviewer’s constructive criticism. Here we would like to explain that, for the following two reasons, we believe our contribution is relevant to deep learning: (i) Our assumptions are as relaxed as, if not more general than, the ones in the related theoretical works in the deep learning context; (ii) The general phenomena (elimination of sharp minima) studied in our setting still persist even in the general (multi-dimensional) loss landscape with uncountably many, connected local minima. In the following we elaborate on these two points:
>
> **(i) Comparison of the assumptions between our work and the existing literature**
>
> We briefly review the assumptions in the similar works about time spent around and preference to different local minima. When discussing the metastability and preference to flat minima of heavy-tailed dynamics, the seminal work [7] assumes isolated, finitely many local minima. The recent works about path-wise dynamics or exit times make very specific assumptions on the loss functions (finitely many isolated local minima; exactly quadratic loss; mean squared loss; etc.) and/or require the noise covariance to be of very specific form (being isotropic OR equal to Hessian of the loss); see [1,2,4,6,14] (where [1,2] are mentioned by the reviewer as well). Some works assume strong convexity and analyze the first exit time from a single region where strong convexity holds; see [15]. In short, we believe that the setting of our work is not more restrictive than those in the latest literature on this topic, not to mention that our work reveals a rich mathematical structure in the first exit times and elimination of sharp minima in truncated heavy-tailed SGD that has not been characterized by existing works.
>
> **(To Be Continued in the Comment Below)**

---

### Official Review · Reviewer_kP7Y · 2021-10-27

**Correctness:** 3
**Technical Novelty And Significance:** 4
**Empirical Novelty And Significance:** 3
**Recommendation:** 8
**Confidence:** 5

**Main Review:**

### Pros

- A beautiful theoretical analysis is provided for a one-dimensional landscape under some structure assumptions.
- Very insightful synthetical experiments are provided to justify the intuitions and theory.
- Inspired by the above analysis, the authors proposed two variants of SGD with injected heavy-tailed noise + gradient clipping. These modified SGDs are expected to converge to flatter minima, thereby generalizing better.
- The experiments for deep nets on (Fashion)MNIST and CIFAR-10/CIFAR-100 are sufficient and show promising improvements over vanilla SGD.

### Cons

It seems that all the analyses ignore the noise structure and are only concerned with the magnitude. For instance, for even a one-dimensional problem, the gradient noise of vanilla mini-batch SGD is state-dependent. However, gradient noises in the prototypical dynamics analyzed in this paper (see Eq. (2)) are iid random variables. As another example, Figure 1 can be misleading since the light-tailed SGDs only use state-independent noise. It is possible that SGD with structured noise can avoid those sharp minima (completely). The authors should disentangle the effect of noise magnitudes and noise structures, and state clearly what is concerned in this paper.



For the related work, previous work studying the stability-driven escaping from sharp minima is completely ignored. I would suggest comparing with them (see [1,2,3] and the reference therein). For example, the stability-driven escaping can tell us that, in Figure 1, GD with a relatively large learning rate never converge to the sharp minima: m1, m2.


### Other comments

- In Figure 1, please make it clear in the caption that where the SGD starts from.

- In the second paragraph of page 2, I am not sure why SGD with light-tailed noise never escape the sharp minima m3. I suppose that by adding large enough noise, SGD at least can escape from it, although it is as efficient as the heavy-tailed SGD.

- " if we stop training SGD at an arbitrary time point, it is almost guaranteed that it won’t be at a sharp minimum". This claim seems wrong to me. Although heavy-tailed SGD with gradient clipping can avoid the sharp minima completely, it may take a very long time.

- In Section 4, "Contrary to the report in (S ̧ims ̧ekli et al., 2019a), heavy-tailed noise may not be ubiquitous in image classification tasks.". Why is there this contradiction? In particular, Table 1 suggests that SGD+heavy-tailed noise performs very badly, even worse than large-batch SGD. Can you explain it? This also contradicts the synthetical experiments, where the heavy-tailed SGD converges to flat minima more likely than light-tailed SGD.

- In paragraph above Section 5, "event"-> "even".



[1] Wu, Lei, et al.,  How sgd selects the global minima in over-parameterized learning: A dynamical stability perspective.

[2] Jastrzebski et al. The Break-Even Point on Optimization Trajectories of Deep Neural Networks.

[3] Cohen, et al. Gradient Descent on Neural Networks Typically Occurs at the Edge of Stability

**Summary Of The Paper:**

This paper study the long-time behavior of heavy-tailed SGD with gradient clipping. It is found that gradient clipping is crucial for heavy-tailed SGD to avoid sharp minima. The basic intuition is that the clipping operation reduces the distance moved by each SGD update. Therefore, for minima narrow than the threshold, the clipping does not change the first exit time. However, for wide minima, SGD is slowed down and takes more time to escape. Consequently, it is more likely that SGD locates in wide minima.



**Summary Of The Review:**

This paper reveals that heavy-tailed noise+gradient clipping can help SGD eliminate the sharp minima. Beautiful theoretical analysis and insightful numerical experiments are provided.

---

> ### Author Response · Authors · 2021-11-24
> **Authors’ Response to Reviewer kP7Y (5/5)**
>
> **(Continuing the Above Comment)**
>
>
>
> In summary, without gradient clipping, heavy-tailed SGDs have a tendency to drift toward the boundary (or infinity), which is present in both the theory and our synthetic experiments. Such bias toward the boundary, however, is more pronounced in high-dimensional deep learning settings, and hence, along with its hypersensitivity to parameter choices, these effects seem to offset heavy-tailed SGD’s enhanced ability to escape from local minima (compared to that of the light-tailed SGDs), leading to deteriorated generalization performance. On the other hand, when gradient clipping is employed, such bias and sensitivity are mitigated (if not completely removed) from the heavy-tailed SGDs, and hence, the desired impact of the truncated heavy tails materialize in the experimental results as theory predicts.
>
>
>
> [1] Wu, Lei, et al., How sgd selects the global minima in over-parameterized learning: A dynamical stability perspective.
>
> [2] Jastrzebski et al. The Break-Even Point on Optimization Trajectories of Deep Neural Networks.
>
> [3]  Cohen, Jeremy, et al. Gradient Descent on Neural Networks Typically Occurs at the Edge of Stability
>
> [4] Abhishek Panigrahi, Raghav Somani, Navin Goyal, and Praneeth Netrapalli. Non-gaussianity of stochastic gradient noise.
>
> [5] Garg, Saurabh, et al. On Proximal Policy Optimization's Heavy-tailed Gradients.
>
> [6] Zhang, Jingzhao, et al. Why are adaptive methods good for attention models?
>
> [7] Pavlyukevich, Ilya. Metastable Behaviour of Small Noise Lévy-Driven Diffusion.
>
> [8] Srinivasan, Vishwak, et al. Efficient Estimators for Heavy-Tailed Machine Learning.
>
> [9] Xie, Zeke, Issei Sato, and Masashi Sugiyama. A diffusion theory for deep learning dynamics: Stochastic gradient descent exponentially favors flat minima.
>
> [10] Şimşekli, Umut, et al. On the Heavy-Tailed Theory of Stochastic Gradient Descent for Deep Neural Networks.
>
> [11] Bovier, Anton, and Frank Den Hollander. Metastability: a potential-theoretic approach
>
> [12] Clauset, Aaron, Cosma Rohilla Shalizi, and Mark EJ Newman. Power-law distributions in empirical data.

---

> ### Author Response · Authors · 2021-11-24
> **Authors’ Response to Reviewer kP7Y (4/5)**
>
> **(Continuing the Above Comment)**
>
> **(ii) Why does the inflated tail in the gradient noise without truncation deteriorate the generalization performance of the vanilla SGD?**
>
> Next, we explain the deteriorated generalization performance when heavy-tailed gradient noises are injected without gradient clipping. We believe that the reason is two-fold:
>
> Technically, the unclipped heavy-tailed SGD does **not** strictly prefer flatter minima, and there is an “implicit drift” towards local minima that are on the boundary.
> Unclipped heavy-tailed SGD seems to be much more sensitive to the choices of hyperparameters than the other training schemes.
>
> Regarding the first point, let us start with mentioning that the arguments about the heavy-tailed (but without gradient clipping) SGDs’ preference for wider basins are based on the results in [7], where the multiplicative constant of the first exit time of the unclipped heavy-tailed SGD was characterized by the width of the basin. (Here we stress that only the multiplicative constants are affected by the width, and the rates are the same for all the basins.)  However, the long-run proportion of the time spent at different minima (i.e. stationary distribution for the Markov chain in Theorem 1 of [7]) is strongly affected by how often (or how easy) a basin can be reached from outside as well. For example, in Figure 2 (Left) of our work, to reach a basin that is *not* leftmost or rightmost (in this case, $\Omega_2$) from $m_3$, the size of the jump has to fall in a specific range (in this case, $[m_3 - s_2, m_3 - s_1]$). In comparison, to reach $\Omega_1$, any leftward jump larger than $m_3 - s_1$ would suffice, even though $m_1$ is considered narrow due to the short escape length $s_1 - m_1$. (In fact, during our experiments, such overshoot and sudden leaps towards the boundary were observed quite often in the heavy-tailed SGDs when gradient clipping is not employed.) Moreover, even if we only consider the first exit time, one can see that the minima on the boundary are naturally preferred when gradients are not clipped, due to the fact that they can only be exited from one side (see the $\lambda_i$ functions in Proposition 5.2 and the $q_i$ constants in Theorem 1 of [7]). In summary, for loss landscapes that are more complicated than the simple two-basin case in Figure 3 of [10], the stationary distribution will be strongly affected by placements/locations of the basins and the tail of noise distribution, so they won’t always accurately represent how “wide” each basin is. This pattern is also observed in our experiments. In Figure 1 (Left, a), one can see that unclipped heavy-tailed SGD stays much longer at sharp minimum $m_1$ when compared to $m_3$ simply because $m_1$ is the leftmost local minimum and on the boundary.
>
> On the other hand, as we rigorously establish in Theorem 1 and Theorem 2 of our paper, when gradient clipping is employed, there is a difference in the *rates* (rather than just multiplicative constants) of the exit times depending on the relative width $l^*$. This translates to much sharper preference for wider minima. Besides, gradient clipping immediately precludes extremely large leaps in a single iteration preventing drift to the boundary.
>
> Regarding the second point, we would like to mention that, compared to all the other methods used in the ablation study,  SGD + Heavy-tailed noises (without clipping) was the only one that required significant efforts on tuning to achieve reasonable accuracy/loss. In fact, under the default choices stated in the paper, accuracy for the unclipped heavy-tailed SGD method tends to suddenly drop and then stay at a random level, even though we always clip the model weights to prevent its norm from exceeding some prefixed threshold $L$. During the tuning process, we were only able to improve its performance by choosing a significantly smaller learning rate $\eta$, possibly because it makes extremely large jumps less often. However, this also resulted in a much longer training time, hence more challenges in fine-tuning. In comparison, the clipped heavy-tailed SGD method doesn’t require much tuning. In fact, the effects of tuning $\alpha$ or the clipping threshold seem marginal in our experiments. In view of this, we leave open the possibility that unclipped heavy-tailed SGD can show better performance (than what we presented in the paper) under suitable hyperparameters, but the method appears to be very sensitive to the parameter choices especially when compared to the truncated heavy-tailed SGD.
>
> Lastly, we’d like to mention that the performance deterioration under unclipped heavy-tailed noises has also been reported in existing literature in various settings (see, for instance, [5,6,8]), and our deep learning results regarding unclipped heavy-tailed SGDs are aligned with these observations.
>
> **(To Be Continued in the Comment Below)**

---

> ### Author Response · Authors · 2021-11-24
> **Authors’ Response to Reviewer kP7Y (3/5)**
>
> **(Continuing the Above Comment)**
>
>
>
> **Q: In Section 4, "Contrary to the report in [10], heavy-tailed noise may not be ubiquitous in image classification tasks.". Why is there this contradiction? In particular, Table 1 suggests that SGD+heavy-tailed noise performs very badly, even worse than large-batch SGD. Can you explain it? This also contradicts the synthetical experiments, where the heavy-tailed SGD converges to flat minima more likely than light-tailed SGD.**
>
>
> Thank you for asking these important questions, which we answer in two parts: (i) why do we claim that there are no heavy tails in the gradient noises in our baseline experiments; and (ii) why does the injected heavy-tail without truncation deteriorate the generalization performance of the vanilla light-tailed SGD.
>
> **(i) Evidence of the light-tailed gradient noises in our baseline experiments**
>
> First, regarding the gradient noise distributions in the benchmark image classification tasks, while heavy tails in gradient noises have been reported in the seminal work [10], we stress that estimation of the heavy-tail index is highly sensitive to the distributional assumptions on the data and the estimation methods applied. For instance, [10] works with the strong assumption that gradient noise at each coordinate is an iid sample from the same symmetric $\alpha$-stable distribution; as a result, when estimating the tail index, the inputs are values at each coordinate of the gradient noise vector (instead of the norm/size of the entire gradient noise) and the method is tailored for $\alpha$-stable distributions. When the coordinate-wise independent $\alpha$-stable assumption is not in force, [5,6] report that it often becomes impossible to detect power-law tails in gradient noises, which is consistent with what we found in our experiments. Please see Appendix K (the last section in the appendix) of the *revised version* of our paper for the tail analysis of the gradient noises in our experiments. In view of the well-established wisdom in heavy-tail literature that there is no single perfect tail estimator, we analyzed the stochastic gradient noise with four different methods: QQ plot, empirical mean residual life (EMRL), Hill plot, and PLFIT [12]. We applied these estimation/diagnostic tools (i) at the beginning of the training, (ii) halfway through the training, (iii) at the end of the training. Throughout all our experiments, we consistently observed strong evidence that the gradient noises are light-tailed, and even if (against all odds) the noises were from a heavy-tailed distribution, the tail index should be far greater (hence, the resulting tail is much lighter) than the heavy-tails we inject (or the popular alpha stable assumption), and hence, the point we make with our tail-inflation experiment is still valid. For the more detailed discussion of the tail analysis results in Appendix K, please see our response to Reviewer QXyo:
>
> Part 1/3: https://openreview.net/forum?id=B3Nde6lvab&noteId=M1zBI-93w8z
>
> Part 2/3: https://openreview.net/forum?id=B3Nde6lvab&noteId=RRYAde4Cnco
>
> Part 3/3: https://openreview.net/forum?id=B3Nde6lvab&noteId=4R117rc9rW7
>
>
> **(To Be Continued in the Comment Below)**

---

> ### Author Response · Authors · 2021-11-24
> **Authors’ Response to Reviewer kP7Y (2/5)**
>
> **(Continuing the Above Comment)**
>
> **Q: In Figure 1, please make it clear in the caption that where the SGD starts from. In paragraph above Section 5, "event"-> "even".**
>
> Thanks for pointing these out. We have fixed them in the revised version of the paper.
>
>
> **Q: In the second paragraph of page 2, I am not sure why SGD with light-tailed noise never escapes the sharp minima m3. I suppose that by adding large enough noise, SGD at least can escape from it, although it is not as efficient as the heavy-tailed SGD.**
>
> If one waits for a long enough time, SGD will eventually escape from any local minimum regardless of whether it is sharp or flat and light-tailed or heavy-tailed. However, the amount of time it takes for an SGD to escape from a local minimum can be dramatically different depending on the tail distributions. In heavy tailed cases, the first exit time grows at a polynomial rate as the learning rate becomes smaller; in light-tailed cases, on the other hand, the first exit time grows at an exponential rate. In our experimental setting under which Figure 1 was produced, the classical (light-tailed) metastability theory—see, for example, (2.1.2) in [11]—predicts that it takes about $10^{450}$ time steps for light-tailed SGDs to escape from the local minimum $m_3$. Therefore, the simulation outcomes for the light-tailed SGDs in Figure 1 are as expected.
> As pointed out by the reviewer, if we increase the magnitude of the gradient noises or increase the learning rate, the SGD will be able to exit from local minima earlier. For example, if we increase the learning rate from $\eta = 0.001$ to $\eta = 0.01$, the first exit time will be of order $10^{24}$ instead of $10^{450}$, and the expected exit time will be less than 100 if the learning rate is $\eta=0.1$. That being said, it should be noted that if the noise magnitude and the learning rate are not small enough, then the noise will dominate the gradient signal in SGD’s dynamics, and SGD will then resemble random walks rather than gradient descents. As a result, the dynamics of the SGD with such hyper-parameters won’t be very useful for the purpose of training (i.e., optimizing the loss function).
>
>
> **Q: " if we stop training SGD at an arbitrary time point, it is almost guaranteed that it won’t be at a sharp minimum". This claim seems wrong to me. Although heavy-tailed SGD with gradient clipping can avoid the sharp minima completely, it may take a very long time.**
>
> Thank you for pointing out the ambiguity in our writing here. What we meant is that “if we stop training SGD at an arbitrary time *after $t/\eta^\beta$ or more steps*, it is almost guaranteed that SGD won’t be at a sharp minimum.” Although the exact number of time steps needed for this depends on the clipping threshold and the underlying loss landscape (which are encoded in $\beta$), we emphasize that the polynomial scale $t/\eta^\beta$ is in stark contrast to the light-tailed counterpart, where the corresponding time scale is exponential w.r.t. $\eta$. For reasonably small $\eta$’s, such an exponential time scale associated with the light-tailed case can quickly become astronomically long (for example, when it comes to the light-tailed SGD in Figure 2, the classical Eyring-Kramers bound suggest that the required time scale will be of order $10^{450}$; see [11] for example), whereas, in the heavy-tailed case, $t/\eta^\beta$ is often well within realistic training time horizons. We will make what we meant clear in the final version of our paper.
>
>
> **(To Be Continued in the Comment Below)**

---

> ### Author Response · Authors · 2021-11-24
> **Authors’ Response to Reviewer kP7Y (1/5)**
>
> We thank reviewer kP7Y for insightful comments and questions. Please see our responses below:
>
> **Q: It seems that all the analyses ignore the noise structure and are only concerned with the magnitude. For instance, for even a one-dimensional problem, the gradient noise of vanilla mini-batch SGD is state-dependent. However, gradient noises in the prototypical dynamics analyzed in this paper (see Eq. (2)) are iid random variables. As another example, Figure 1 can be misleading since the light-tailed SGDs only use state-independent noise. It is possible that SGD with structured noise can avoid those sharp minima (completely). The authors should disentangle the effect of noise magnitudes and noise structures, and state clearly what is concerned in this paper.**
>
> We thank the reviewer for pointing this out. In the final version of our paper, we will make it clear that we focus on the effect of the tail distribution of the gradient noises while assuming a state-independent noise structure. Specifically, we will add the following sentence to Section 1:
> “While in full generality the structure of gradient noises in SGD is state-dependent, in this work we focus on the role of noise magnitude and analyze the setting where each SGD update is perturbed by iid heavy-tailed noise. ”
> (We did not add the above sentence to the current version of our paper yet due to the page limit. We will add it to the final version, which we believe will have an extra page.)
>
> On the other hand, we would also like to mention that the key proof ideas for the current results can be extended to the cases with general state-dependent noise through more involved bookkeeping. That is, if we assume that the noise at state $x$ is $\sigma(x)Z$ where $Z$ is a heavy-tailed RV, and the function $\sigma(\cdot)$ captures the state-dependence of the noise structure, then a similar result to Theorem 1 can be proved with constants $q_i, q_{i,j}$ adjusted accordingly. Specifically, in eq. (E.2) and (E.4) in Appendix, the terms $w_1$ and $w_j$ need to be multiplied by $\sigma\big(\hat{x}(t-)  \big)$, i.e. the noise structure right before the large jump. This would furthermore affect the quantities defined in eq. (E.5) and (E.6), and hence the $q_i, q_{i,j}$ in (E.7) as well. With these modifications, our techniques would carry over to the state-dependent setting.
>
>
> **Q: For the related work, previous work studying the stability-driven escaping from sharp minima is completely ignored. I would suggest comparing with them (see [1,2,3] and the reference therein). For example, the stability-driven escaping can tell us that, in Figure 1, GD with a relatively large learning rate never converge to the sharp minima: m1, m2.**
>
> We thank the reviewer for pointing us to these works on the stability of SGD. We will include these references in Section 1 of the final version of our paper.
>
> [1] takes the perspective of linear stability and establishes conditions for SGD to get attracted to or avoid certain solutions based on learning rate, batch size, and the concept of non-uniformity of local minima. Inspired by [1], [2] analyzes the trajectory-wise stability of SGD and found that a “break-even point” partitions the training procedure into two phases: the implicit regularization effects in SGD due to a larger learning rate becomes visible in the second phase after this “break-even point”. Similarly, [3] reports that typical GD trajectories in standard image classification tasks can be partitioned into two phases: in the first “progressive sharpening” phase the sharpness of the Hessian monotonically increases, while in the second phase we observe the “edge of stability” regime where the sharpness of Hessian hovers slightly above the critical value $2/\eta$ and the training loss slowly decrease in an oscillating fashion.
>
> In comparison, our results provide a different perspective (which complements those of the above mentioned works) by showing that, under heavy-tailed noises, SGD (especially with gradient clipping) exhibits a different type of algorithmic regularization that is not revealed in [1,2,3]. In particular, one major difference is that the algorithmic regularization we analyze persists even when the learning rate converges to zero.
>
> We will include a detailed discussion of this line of works ([1,2,3] and references therein) and comparisons to our work in the appendix of our paper.
>
> **(To Be Continued in the Comment Below)**

---

### Official Review · Reviewer_QXyo · 2021-11-03

**Correctness:** 3
**Technical Novelty And Significance:** 3
**Empirical Novelty And Significance:** 3
**Recommendation:** 6
**Confidence:** 3

**Main Review:**

I believe that the topics of SGD-trained networks' generalization in general, and the role of heavy-tailed parameter statistics in this in specific are very timely and worthy of attention, since our understanding of the dynamics that lead to generalization is not on par with the empirical success of SGD-based methods.

I find the authors' analyses and results interesting and well-presented. I think it has potential to improve our understanding of the generalization characteristics of SGD-trained networks. Although their analyses mostly focus on the univariate case, I believe that this is acceptable as they provide some seminal results regarding truncated-gradient SGD.

However, I have a hard time understanding the authors' characterization of recent results in the literature (or lack thereof), which also informs their experiment design. My point can be most dramatically made by drawing attention to the choice of baseline that the authors present. While the recent theoretical and empirical findings in literature emphasize the relationship between learning rate, and tail index and/or generalization, the authors somehow base their methodology and experiments on the supposed absence of heavy tails in gradient noise in image classification tasks. This is not necessarily true, and this fact is well-documented. Given this fact, the fact that the authors present the baselines as thin-tailed noise algorithms (without any detailed analysis of learning rate / batch size and their effects on generalization), as well as not estimating the tail index of these noises is surprising. I think the authors need to take into account more recent results in the literature and possibly alter their experimental settings and discussion accordingly. This is especially important since the authors aim to analyse why a specific modification of SGD leads to improvements.

- Hodgkinson, Liam, and Michael W. Mahoney. 2020. “Multiplicative Noise and Heavy Tails in Stochastic Optimization.” ArXiv:2006.06293 [Cs, Math, Stat], June. http://arxiv.org/abs/2006.06293.
- Gurbuzbalaban, Mert, Umut Şimşekli, and Lingjiong Zhu. 2021. “The Heavy-Tail Phenomenon in SGD.” ArXiv:2006.04740 [Cs, Math, Stat], June. http://arxiv.org/abs/2006.04740.
- Lewkowycz, Aitor, Yasaman Bahri, Ethan Dyer, Jascha Sohl-Dickstein, and Guy Gur-Ari. 2020. “The Large Learning Rate Phase of Deep Learning: The Catapult Mechanism,” March. https://arxiv.org/abs/2003.02218v1.

**Summary Of The Paper:**

The authors analyse the behavior of SGD under gradient clipping. Their analysis in the univariate case shows that gradient clipping in the heavy-tailed gradient noise (almost) eliminates the algorithm's tendency to stay at sharp minima. The authors support their analysis with synthetic experiments. The authors then conduct experiments on real data where they add heavy tailed noise to the gradient, and clip it afterwards.

**Summary Of The Review:**

The authors present interesting analyses and results regarding a modified version of SGD in optimization. Their characterization of the recent literature, and the experimental design based thereupon seems to need more attention.

---

> ### Author Response · Authors · 2021-11-23
> **Authors’ Response to Reviewer QXyo: References**
>
> [1] Hodgkinson, Liam, and Michael W. Mahoney. Multiplicative Noise and Heavy Tails in Stochastic Optimization.
>
> [2] Gurbuzbalaban, Mert, Umut Şimşekli, and Lingjiong Zhu. The Heavy-Tail Phenomenon in SGD.
>
> [3] Lewkowycz, Aitor, Yasaman Bahri, Ethan Dyer, Jascha Sohl-Dickstein, and Guy Gur-Ari. The Large Learning Rate Phase of Deep Learning: The Catapult Mechanism.
>
> [4] Şimşekli, Umut, et al. On the Heavy-Tailed Theory of Stochastic Gradient Descent for Deep Neural Networks.
>
> [5] Xie, Zeke, Issei Sato, and Masashi Sugiyama. A diffusion theory for deep learning dynamics: Stochastic gradient descent exponentially favors flat minima.
>
> [6] Abhishek Panigrahi et al. Non-gaussianity of stochastic gradient noise.
>
> [7] Alstott, Jeff, Ed Bullmore, and Dietmar Plenz. powerlaw: a Python package for analysis of heavy-tailed distributions.
>
> [8] Garg, Saurabh, et al. On Proximal Policy Optimization's Heavy-tailed Gradients.
>
> [9] Zhang, Jingzhao, et al. Why are adaptive methods good for attention models?
>
> [10] Resnick, Sidney I. Heavy-tail phenomena: probabilistic and statistical modeling. Springer. 2007.
>
> [11] Clauset, Aaron, Cosma Rohilla Shalizi, and Mark EJ Newman. Power-law distributions in empirical data.
>
> [12] Keskar, Nitish Shirish, et al. On large-batch training for deep learning: Generalization gap and sharp minima
>
> [13] Zhu, Zhanxing, et al. The anisotropic noise in stochastic gradient descent: Its behavior of escaping from sharp minima and regularization effects.
>
> [14] Smith, Samuel L., et al. Don't decay the learning rate, increase the batch size

---

> > ### Comment · Reviewer_QXyo · 2021-11-29
> > **Thanks for the response**
> >
> > I thank the authors for their detailed response to my questions and concerns.
> >
> > Through their comments I was able to better understand the major point of emphases in their paper (which I assume will also be improved in the camera-ready version of the paper), as well as how they thought their findings compared to other significant findings.
> >
> > Given the fact that the points raised above (which were at least partly shared by the other reviewers) required elaborate discussion implies that the authors also should modify their main paper to present a more discerning presentation of these points, using the extra space afforded if accepted. More specifically, a more nuanced discussion of the following points would be appreciated: 1- the dependence of the emergence of heavy tailed noise on hyperparameters/architecture/dataset (which does not invalidate their analyses and results), 2- how findings regarding the heavy-tailed nature of parameters and gradient noises should be understood with respect to the current paper. Obviously, the authors cannot include the whole discussion in the main paper but should definitely include the gist of it in the main text and refer the interested reader to the appendix.
> >
> > In light of authors' comments and discussion, I raise my score to recommend the acceptance of the paper.

---

> > > ### Author Response · Authors · 2021-11-30
> > > **Thanks for the response**
> > >
> > > We thank the reviewer for the positive response. We agree that the main paper should present the gist of this discussion. We will make the suggested changes to the final version of our paper.

---

> ### Author Response · Authors · 2021-11-23
> **Authors’ Response to Reviewer QXyo (3/3)**
>
> **(Continuing the Above Comment)**
>
> Regarding the literature that reports the heavy-tails in the gradient noises in image classification tasks (see, for instance, [4]), we make it clear that our experiments above are *not* designed to prove or disprove them. Rather, our point is that under the choice of reasonable hyperparameters (adopted from the literature [12,13,14]), we found strong evidence that the gradient noises are light-tailed. Although we didn’t manage to find them in our experiments, we leave open the possibility that there are heavy-tails naturally arising in the gradient noise under different choices of hyperparameters. That being said, we again stress that the estimation of the heavy-tail index is particularly sensitive to the distributional assumptions on the data and the estimation methods applied; specifically, [4] works with the strong assumption that gradient noise at each coordinate is an iid sample from the same symmetric $\alpha$-stable distribution. As a result, when estimating the tail index, the inputs are values at each coordinate of the gradient noise vector instead of the actual norm/size of the gradient noise; moreover, the tail-index estimation method in [4] is tailored for $\alpha$-stable distributions. When the coordinate-wise independent $\alpha$-stable assumption is not in force, [5,6] report that it often becomes impossible to detect power-law tails in gradient noises, which is consistent with our experiment results.
>
> In summary, we believe that power-law heavy tails may be present (or even prevalent) in other contexts such as reinforcement learning and attention models (see [8,9]), but not necessarily in image classification tasks tested in our paper. Instead, extensive statistical analyses in the Appendix K suggest the absence of heavy tails in the gradient noises in our experiments. Therefore, the characterization of the vanilla SGD as the light-tailed (or at least lighter than the inflated tails) benchmark in our experiments is valid, and our ablation study is well grounded.
>
>
> **(ii) Recent theoretical results that suggest power-law tails and their relation to learning rate / batch size does not contradict our baseline model:**
>
> Returning to the references mentioned by the reviewer, we reiterate that what has been established in [1,2] is that the **stationary distribution of the SGD** can be heavy-tailed. This, however, does not imply that the **gradient noise** is heavy-tailed. In particular, the heavy-tail index (or heavy-tailedness itself) of the gradient noise is not affected by the learning rate and the batch size as we elaborate below. Indeed, in all our experiments, we could not find heavy-tailed gradient noise regardless of the choice of learning rates and batch sizes we tried. Therefore, the premise (light-tailed baseline) of our experiment is not inconsistent with [1,2].
>
> To be specific, [1,2] show that heavy tails can arise in the stationary distribution of SGD through multiplicative dynamics, and the tail index of the resulting stationary distribution can be characterized by the learning rates and the magnitude of noises (through batch size). However, the results in [1,2] do not imply the existence of heavy tails in the **gradient noises**. In both papers (as well as other works in the literature), the “heaviness” of the tail of the gradient noise (e.g., power-law index $\alpha$ in heavy-tailed cases) is fixed in the model (same as in our setting) and not entangled with the learning rate or batch size. For example, $B_k$ in (5) of [1] corresponds to the gradient noise, and its tail index doesn’t depend on the learning rate or the batch size. **In particular, the change of learning rate does not induce heavy tails in $B_k$**.  (Again, what is entangled with the learning rate and the batch size is the power-law index of the **stationary distribution** of the SGD, not the power-law index of the **gradient noise distribution**). On the other hand, we focus on the impact of the heavy-tails **in the gradient noise** and the truncation scheme for (any) fixed batch size and small learning rates on the global dynamics of SGD. In view of this, it should be clear that our analysis can be decoupled from the choice of batch size or the impact of the learning rate on SGD’s stationary distribution. Therefore, our observation and the design of the experiments are compatible with the references mentioned by the reviewer.
>
> **(For List of References, See Comment Below)**

---

> ### Author Response · Authors · 2021-11-23
> **Authors’ Response to Reviewer QXyo (2/3)**
>
> **(Continuing the Above Comment)**
>
> **QQ plots** (Fig. K.1 - K.15) clearly show that the tails in noise distribution are always much lighter than the Pareto distributions with $\alpha = 2$ or even 10. Therefore, the typical power-law assumption, especially the alpha-stable distributions in [4] (with $\alpha \in (0,2)$ ), seems far from the distribution of the actual data we obtained in the image classification tasks. In fact, the tail of the noise distributions seems to be between that of lognormal and normal distributions, implying that it is lighter than any power-law distribution.
>
> In Figure K.16-K.20 we plotted the **empirical mean residual life (EMRL)** of the gradient noise distributions. It is well known that the mean residual life blows up to infinity if and only if the distribution is heavy-tailed (more precisely, long-tailed). However, from the figures, one can see that none of the EMRL exhibits such a pattern in any case tested in our experiments. Instead, we see clear downward trends, which strongly suggest light tails, in all the tested cases.
>
> If we assume a power-law tail, the **Hill estimator** is a classical tail index estimator with a long history in extreme value theory literature. A critical algorithmic parameter of the Hill estimator is the number of order statistics used in the estimation, and the Hill plot is a popular exploratory tool for investigating the Hill estimators with different numbers of order statistics. Although it is well known that Hill estimators and Hill plots are fallible if the power-law assumption is not satisfied, (and hence, it is not suited for deciding whether a given set of samples are from light-tailed distribution or heavy-tailed distribution; in particular, the method will return some power-law tail index $\alpha$ whether or not the samples are from a heavy-tailed distribution or a light-tailed one) we present the Hill plot to see what would be the estimated tail indices if the gradient noises hypothetically followed a power law. In the Hill plots shown in Figure K.21-K.25, we presented the rescaled version (altHill) of the Hill plots (see Chapter 4.4 in [10]) for the following reason. Hill estimator is a consistent estimator of the power-law index when the proportion of the samples used approaches 0, and altHill plots allow us to scrutinize the estimated indices under a small proportion of samples. In particular, the points around the red dashed lines correspond to estimation using the top 1% of the samples. We can see that most Hill plots stay well above 10 for the most part and almost never drop below 2. This strongly suggests that even if the gradient noises are from a heavy-tailed distribution, it is likely to have a very high power-law index (implying relatively lighter tails), and hence, we cannot expect to observe a prominent heavy-tailed behavior from them.
>
> A popular data-driven approach with statistical guarantees (again, under the assumption that the samples are indeed from a heavy-tailed distribution) to select the number of order statistics in the Hill plot is **PLFIT** [11]. We **estimated the power-law indices** using the python implementation [7] of PLFIT. The numbers are presented in Table K.1. All the estimations are at least 5 for all cases tested in our experiments, and most of the time, the estimation is above 10. Again, this means that even under the hypothetical assumption (against what the QQ plots and EMRLs suggest) that the gradient noises were from a heavy-tailed distribution, the tail indices of the gradient noises should be large, and hence, the gradient noises in our experiments have much lighter tails than any $\alpha$-stable distribution (which requires $\alpha<2$) or the heavy-tailed noises we injected during tail inflation experiments ($\alpha = 1.4$).
>
> **(To Be Continued in the Comment Below)**

---

> ### Author Response · Authors · 2021-11-23
> **Authors’ Response to Reviewer QXyo (1/3)**
>
> We thank reviewer QXyo for thoughtful comments and questions. We believe that the raised questions are important, and our paper would benefit from further clarification. Hence, we included (the gist of) the following discussion in this comment in Appendix A.5 of the revised paper.
>
>
> **Reviewer’s Comment:**
>
> *However, I have a hard time understanding the authors' characterization of recent results in the literature (or lack thereof), which also informs their experiment design. My point can be most dramatically made by drawing attention to the choice of baseline that the authors present. While the recent theoretical and empirical findings in literature emphasize the relationship between learning rate, and tail index and/or generalization, the authors somehow base their methodology and experiments on the supposed absence of heavy tails in gradient noise in image classification tasks. This is not necessarily true, and this fact is well-documented. Given this fact, the fact that the authors present the baselines as thin-tailed noise algorithms (without any detailed analysis of learning rate / batch size and their effects on generalization), as well as not estimating the tail index of these noises is surprising. I think the authors need to take into account more recent results in the literature and possibly alter their experimental settings and discussion accordingly. This is especially important since the authors aim to analyse why a specific modification of SGD leads to improvements.*
>
>
> **Authors’ Response:**
>
> We thank the reviewer for asking this question and sharing references [1,2,3]. Indeed, the learning rate and the batch size do affect the generalization performance of SGD (as argued in [3] and many others in the literature), and the recent theoretical analyses (such as [1,2]) of training algorithms characterize the relationship between the hyperparameters (such as learning rate and batch size) and the heavy-tails that arise in the stationary distribution of SGD. In fact, one of the first goals we set in our deep learning experiments was to confirm the heavy tails in the stochastic gradient noise (to design an ablation study on the effect of clipping accordingly); however, we didn’t manage to confirm heavy tails in the gradient noise throughout our experiments with the image classification problems.
>
> While such absence of heavy-tails was unexpected, we believe that *it allowed us to perform a more thorough ablation study*, because we were able to devise a method to inflate (from a light tail to a heavy tail) the tail of the gradient noise without introducing bias, whereas it is unclear how to deflate (from a heavy tail to a light tail) the tail of the gradient noise without introducing bias. On the other hand, we emphasize that *our empirical observation and characterization of the baseline experiments are **not contradictory** to the aforementioned theoretical arguments in [1,2]*. In the rest of this comment, (i) we present the empirical evidence that supports our characterization of the baseline model—i.e., the absence of heavy tails in the gradient noise—and (ii) clarify that the emergence of heavy-tails in the **stationary distribution** of SGD (argued in [1,2]) does not contradict the observed absence of heavy tails in the **gradient noise** of SGD. This allows us to study the impact of truncated heavy-tails in the gradient noise separately from the choice of hyper-parameters.
>
>
> **(i) Tail index of noises in our experiments:**
>
> We start with our observations on the noise distributions. Please see section K (the last section in the appendix) of the *revised version* of our paper. In view of the well-established wisdom in heavy-tail literature that there is no single perfect tail estimator, we analyzed the stochastic gradient noise with four different methods: QQ plot, empirical mean residual life (EMRL), Hill plot, and PLFIT [11]. We applied these estimation/diagnostic tools (i) at the beginning of the training, (ii) halfway through the training, (iii) at the end of the training.
>
> Throughout all our experiments, we consistently observe strong evidence that the gradient noises are light-tailed, and even if (against all odds) the noises were from a heavy-tailed distribution, the tail index should be far greater (hence, the resulting tail is much lighter) than the heavy-tails we inject (or the popular alpha stable assumption), and hence, the point we make with our tail-inflation experiment is still valid. We summarize the results as follows
>
> **(To Be Continued in the Comment Below)**

---

### Decision · Program_Chairs · 2022-01-20

**Decision:**

Accept (Poster)

**Comment:**

Motivated by empirical observations that SGD performed on deep networks converge to regions of flatter loss curvature relative to large or full batch GD, the authors perform a theoretical analysis of trajectories of SGD with the presence of heavy tailed noise. The primary observation of the theory is that heavy tailed noise has a higher probability of "kicking" the current parameters to a new region of the input space, which has some probability of lying in a sharper region. However, it's important to note that in this analysis SGD with heavy tailed noise doesn't stay in the sharp regions, but will eventually be kicked back out of it back to other regions. In a sense, this defines a transition graph which predicts that the steady state distribution should spend some fraction of time in different regions of the input space (and different sharpness) while never "converging" anywhere. This is shown most clearly in Figure 1 top center where the heavy tailed SGD randomly jumps between different regions of the input space throughout the entire training trajectory. Experiments are then run on deep networks showing that heavy-tailed SGD with gradient clipping converges to regions of flatter curvature.

Reviews of the work were generally positive, the theory is well presented and Figure 1 does a solid job demonstrating the main idea. The primary criticism was raised by reviewer HGyL, arguing that the results should be largely irrelevant to deep learning. Most of the debate between this reviewer and the authors centered around whether or not ReLU networks have minima which extend off to infinity. The AC will not dig into the details of the argument. It seems clear, however, that if there were a deep learning workload with heavy tailed noise that the authors results will have some relevancy, though the exact nature of the resulting transition graph may have a complicated dependence on the loss surface. Unfortunately the authors were unable to find a such a workload in image classification (there is some prior work suggesting the NLP models with rare tokens may be a better fit) and so needed to artificially induce heavy tailed noise to test their theory. This is a bit of a limitation, but given the clear writing and interesting experiments as noted by reviewers the work seems worth accepting. The AC strongly urges the authors though to include a more lengthy discussed of Wu. et. al. as that work seems to agree with experiment of the sharpness of stable regions selected by SGD when run on deep models without heavy tailed noise.